# RETHINKING THE PRUNING CRITERIA FOR CONVOLUTIONAL NEURAL NETWORK

## ABSTRACT

Channel pruning is a popular technique for compressing convolutional neural networks (CNNs), and various pruning criteria have been proposed to remove the redundant filters of CNNs. From our comprehensive experiments, we found some blind spots on pruning criteria: (1) Similarity: There are some strong similarities among several primary pruning criteria that are widely cited and compared. According to these criteria, the ranks of filters' *importance* in a convolutional layer are almost the same, resulting in similar pruned structures. (2) Applicability: For a large network (each convolutional layer has a large number of filters), some criteria can not distinguish the network redundancy well from their measured filters' *importance*. In this paper, we theoretically validate these two findings with our assumption that the well-trained convolutional filters in each layer approximately follow a Gaussian-alike distribution. This assumption is verified through systematic and extensive statistical tests.

## 1 INTRODUCTION

Pruning (LeCun et al., 1990; Hassibi & Stork, 1993; Han et al., 2015; He et al., 2019) a trained neural network is commonly seen in network compression. In particular, for neural networks with convolutional filters, channel pruning refers to the pruning of the filters in the convolutional layers. There are several critical factors for channel pruning. **Procedures**. One-shot method (Li et al., 2016): Train a network from scratch; Use a certain criterion to calculate filters' *importance*, and prune the filters which have small *importance*; After additional training, the pruned network can recover its accuracy to some extent. Iterative method (He et al., 2018; Frankle & Carbin, 2019): Unlike One-shot methods, they prune and fine-tune a network alternately. **Criteria**. The filters' *importance* can be definded by a given criterion. From different ideas, many types of pruning criteria have been proposed, such as Norm-based (Li et al., 2016), Activation-based (Hu et al., 2016; Luo & Wu, 2017), Importance-based (Molchanov et al., 2016; 2019a), BN-based (Liu et al., 2017b) and so on. **Strategy**. Layer-wise pruning: In each layer, we can sort and prune the filters, which have small *importance* measured by a given criterion. Global pruning: Different from layer-wise pruning, global pruning sort the filters from all the layers through their *importance* and prune them.

Table 1: The pruned filters' index ordered by the filters' *importance* from given pruning criteria, taking VGG16 (3$^{\text{rd}}$ Conv) and ResNet18 (12$^{\text{th}}$ Conv) as examples. The pruned filters' index (the ranks of filters' *importance*) are almost the same from different pruning criterion and it will lead to the similar pruned structures.

| Criteria | Model | Pruned Filters' Index (Top 8) | Model | Pruned Filters' Index (Top 8) |
|---|---|---|---|---|
| $\ell_1$ | ResNet18 | [111, 212, 33, 61, 68, 152, 171, 45] | VGG16 | [102, 28, 9, 88, 66, 109, 86, 45] |
| $\ell_2$ | ResNet18 | [111, 33, 212, 61, 171, 42, 243, 129] | VGG16 | [102, 28, 88, 9, 109, 66, 86, 45] |
| **GM** | ResNet18 | [111, 212, 33, 61, 68, 45, 171, 42] | VGG16 | [102, 28, 9, 88, 109, 66, 45, 86] |
| **Fermat** | ResNet18 | [111, 212, 33, 61, 45, 171, 42, 68] | VGG16 | [102, 28, 88, 9, 109, 66, 45, 86] |

As one of the simplest and most effective channel pruning criteria, $\ell_1$ pruning (Li et al., 2016) is widely used in the industry. The core idea of this criterion is to sort the $\ell_1$ norm of filters in one layer and then prune the filters, which have a small $\ell_1$ norm. Similarly, there is $\ell_2$ pruning (Frankle & Carbin, 2019; He et al., 2018). Through the study of the distribution of norm, He et al. (2019) demonstrates that these criteria should satisfy two conditions: (1) the variance of the norm of the filters can-

not be too small; (2) the minimum norm of the filters should be small enough. Since these two conditions do not always hold, a new criterion considering the relative *importance* of the filters ($\ell_1$ and $\ell_2$ norm can be seen as one of algorithm which uses absolute *importance* of filters) is proposed (He et al., 2019). Since this criterion uses the Fermat point (*i.e.*, geometric median (Cohen et al., 2016)), we call this method **Fermat** . However, due to the high calculation cost of Fermat point, He et al. (2019) relaxed it and then got another criterion **GM** method. Let $F_{ij} \in \mathbb{R}^{N_i \times k \times k}$ represents the $j^{\text{th}}$ filter of the $i^{\text{th}}$ convolutional layer, where $N_i$ is the number of input channels for $i^{\text{th}}$ layer and $k$ denotes the kernel size of the convolutional filter. In $i^{\text{th}}$ layer, there are $N_{i+1}$ filters. The details of these criteria are shown in Table 2. $\mathbf{F}$ denotes the Fermat point of $F_{ij}$ in Euclidean space. These four pruning criteria are called Norm-based pruning in this paper as they include norm in their design.

In previous works (Luo et al., 2017; Han et al., 2015; Ding et al., 2019; Dong et al., 2017; Renda et al., 2020), including the criteria mentioned above, they usually focused on (a) How much the model was compressed; (b) How much performance was restored; (c) The inference efficiency of the pruned network and (d) The cost of finding the pruned network. However, there is little work to discuss two blind spots about the pruning criteria:

Table 2: Norm-based pruning criteria.

| Criterion | Details of *importance* |
|---|---|
| $\ell_1$ (Li et al., 2016) | $\|F_{ij}\|_1$ |
| $\ell_2$ (Frankle & Carbin, 2019) | $\|F_{ij}\|_2$ |
| **Fermat** (He et al., 2019) | $\|\mathbf{F} - F_{ij}\|_2$ |
| **GM** (He et al., 2019) | $\sum_{k=1}^{N_{i+1}} \|F_{ik} - F_{ij}\|_2$ |

**(1) Similarity: What are the actual differences among these previous pruning criteria?** Using VGG16 and ResNet18 on ImageNet, we show the ranks of filters' *importance* under different criteria in Table 1. It is easy to find that they have almost the same sequence, leading to similar pruned structures. In this situation, the criteria used absolute *importance* of filters ( $\ell_1$,$\ell_2$) and the criteria used relative *importance* of filters (**Fermat**,**GM**) may not be significantly different.

**(2) Applicability: What is the Applicability of these pruning criteria to prune the CNNs?** There is a toy example w.r.t. $\ell_2$ criterion. If the $\ell_2$ norm (regarded as *importance*) of the filters in one layer are 0.9, 0.8, 0.4 and 0.01, according to *smaller-norm-less-informative assumption* (Ye et al., 2018), it's easy to know that we should prune the last filter. However, if the norm are close, like 0.91, 0.92, 0.93, 0.92, it is hard to know which filter should be pruned even though the first one is the smallest. As shown in Fig. 1, taking Wide ResNet28-10 (WRN) as an example, this is a real and existing problem for pruning a network with the $\ell_2$ criterion.

Some similar research methods and opinions on pruning criteria are mentioned in He et al. (2019) and Molchanov et al. (2019a). We make further analysis and observation of them in this paper. Since the criteria in Table 2 are widely cited and compared (Liu et al., 2020b; Li et al., 2020b; He et al., 2020; Liu et al., 2020a; Li et al., 2020a), therefore it's important to study them for the above two issues. In order to rigorously study them: First, we come up with an assumption about the distribution of the parameters of the convolutional filters, called *Convolution Weight Distribution Assumption* (CWDA), with systematic and comprehensive statistical tests (Appendix P) in Section 2. Next, in Section 3 and Section 4, we theoretically verify the

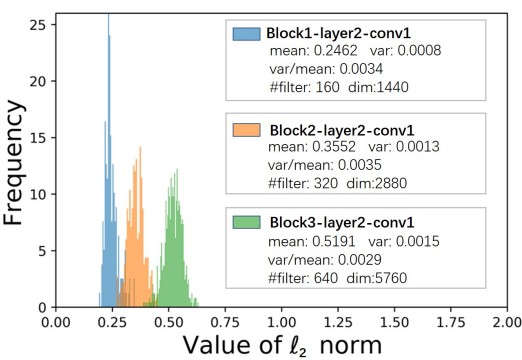

Figure 1: The distribution of $\ell_2$ norm (WRN)

problem about Similarity and Applicability for some Norm-based criteria in layer-wise pruning. Last but not least, in Section 5, we discuss more issues including: (1) the conditions for CWDA to be satisfied, (2) the Similarity and Applicability when we use other types of pruning criteria, (3) and another pruning strategy, the global pruning.

**Contribution**. **(1)** We propose and verify an assumption called CWDA, which reveals that the well-trained convolutional filters approximately follow a Gaussian-alike distribution.

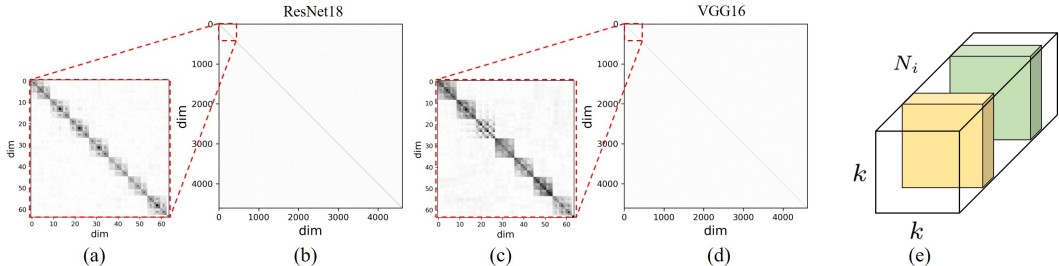

Figure 2: (a-d) Visualization of correlation matrix $FF^T$. More experiments on the different layers of other networks can be found in Appendix Q .(e) The structure of $F_{ij}$.

**(2)** We analyze the Applicability problem and Similarity from different types of pruning criteria. Under CWDA, we strictly validate these issues of Norm-based criteria in layer-wise pruning.

**(3)** Under CWDA, some Norm-based criteria using the global pruning strategy have the problem about the inconsistent magnitude of *importance* between different layers, which explains the phenomenon that they sometimes cut off the network. The estimations using CWDA almost coincides to the actual results obtained from the real network, which also demonstrates the effectiveness of the CWDA.

## 2 WEIGHT DISTRIBUTION-ASSUMPTION

In this section, to study the Similarity and the Applicability problem among the pruning criteria shown in Table 2, we propose and verify an assumption about the distribution of the parameters in convolutional filters.

**(Convolution Weight Distribution Assumption)** Let $F_{ij} \in \mathbb{R}^{N_i \times k \times k}$ be the $j^{\text{th}}$ well-trained filter of the $i^{\text{th}}$ convolutional layer. In $i^{\text{th}}$ layer, $F_{ij}, j = 1, 2, ..., N_{i+1}$ are i.i.d and follow a Gaussian-alike distribution:

$$F_{ij} \sim \mathbf{N}(\mathbf{0}, c^2 \cdot \mathbf{\Sigma}_{\text{block}}), \tag{1}$$

where $c$ is a constant and $\mathbf{\Sigma}_{\text{block}} = \text{diag}(K_1, K_2, ..., K_{N_i})$ is a block diagonal matrix. The values of the off-diagonal elements are close to 0 and $K_l \in R^{k^2 \times k^2}, l = 1, 2, ..., N_i$.

This assumption is based on the observation shown in the Fig. 2. Let $F \in \mathbb{R}^{(N_i \times k \times k) \times N_{i+1}}$ denote all the parameters in $i^{\text{th}}$ layer and we use the correlation matrix $FF^T$ to estimate the shape of $\mathbf{\Sigma}_{\text{block}}$. Taking the last convolutional layers of ResNet18 and VGG16 trained ImageNet as an example, we find that $FF^T$ is a block diagonal matrix. Specifically, each block is a $k^2 \times k^2$ matrix and the off-diagonal elements are close to 0. For $j^{\text{th}}$ filter $F_{ij} \in \mathbb{R}^{N_i \times k \times k}$ in $i^{\text{th}}$ layer as shown in Fig. 2(e), this phenomenon reveals that the parameters in the same channel of $F_{ij}$ tend to be linearly correlated, and the parameters of any two different channels (yellow and green channels in Fig. 2(e)) in $F_{ij}$ only have a low linear correlation. Since the kernel size $k$ is a small constant (like 1 or 3) and $N_i \gg 1$, the length $k^2$ of each block is much smaller than the length $k^2 \cdot N_i$ of the covariance matrix in most convolutional layers. Therefore, the block diagonal matrix $\mathbf{\Sigma}_{\text{block}}$ can be regarded as a diagonal matrix, as shown in Fig. 2(b) and (d). For the convenience of analysis, we relax the assumption, *i.e.*, $F_{ij}$ **approximately** follows a Gaussian-alike distribution[1]:

$$F_{ij} \sim \mathbf{N}(\mathbf{0}, c^2 \cdot \mathbf{I}_{N_i \times k \times k}), \tag{2}$$

where $\mathbf{I}_{N_i \times k \times k}$ is an identity matrix. The statistical tests in Section 2.1 and the estimation by CWDA in Fig. 10 show the reasonability of this relaxation. In the remaining sections, we use Eq.2 to represent CWDA unless otherwise specified. In Fig. 3, taking VGG16 and ResNet18 on ImageNet dataset as examples, we visualize the CWDA through the distribution of the convolutional filters.

---

[1]In Section 5, we make further discussion and analysis on the conditions for CWDA to be satisfied.



Figure 3: Visualization of the distribution of convolutional filters. The parameters of a convolutional filter approximately follow a Gaussian-alike distribution.

## 2.1 STATISTICAL TEST

In fact, CWDA is not easy to be verified. For example, for ResNet164 (On Cifar100), the number of filters in the first stage is only 16, which is too small to be used to estimate the statistics accurately. More other objective reasons are shown in Section 5.1. As these problems, we consider verifying three necessary conditions of CWDA: (1) Gaussian (*i.e.*, to verify whether $F_{ij}$ approximately follows a Gaussian-alike distribution); (2) Standard Deviation (*i.e.*, to verify whether the standard deviation of each filter in any layers is close to a constant $c$); (3) Mean (*i.e.*, to verify whether the mean of $F_{ij}$ is close to 0).

In Table 3, to illustrate that CWDA holds universally, we consider a variety of factors, such as network structure, optimizer, regularization[2], initialization, dataset, training strategy, and other tasks in computer vision (*e.g.*, semantic segmentation, detection, image matting and so on). The details of these statistical tests are shown in Appendix P.

Table 3: The experiments for having the comprehensive statistical tests on CWDA.

| NETWORK STRUCTURE (P.1) | OPTIMIZER (P.2) | REGULARIZATION (P.3) |
|---|---|---|
| ResNet (He et al., 2016a) | SGD (Sutskever et al., 2013) | L1 norm |
| VGG (Simonyan & Zisserman, 2014) | ASGD (Polyak & Juditsky, 1992) | L2 norm |
| AlexNet (Krizhevsky, 2014) | Adam (Kingma & Ba, 2014) | RReLu (Xu et al., 2015) |
| DenseNet (Huang et al., 2017) | Adagrad (Duchi et al., 2011) | Dropact (Liang et al., 2018) |
| PreResNet (He et al., 2016b) | Adamax (Kingma & Ba, 2014) | Autoaug (Cubuk et al., 2019) |
| WRN (Zagoruyko & Komodakis, 2016) | Adadelta (Zeiler, 2012) | Cutout (DeVries & Taylor, 2017) |
| ResNext (Xie et al., 2017) | | Cutmix (Yun et al., 2019) |
| ATTENTION MECHANISM (P.4) | INITIALIZATION (P.5) | DATASET (P.6) |
| SENet (Hu et al., 2018) | Kaiming-normal (He et al., 2015) | CIFAR10 (Krizhevsky et al., 2009) |
| DIANet (Huang et al., 2019) | Kaiming-uniform (He et al., 2015) | CIFAR100 (Krizhevsky et al., 2009) |
| SRMNet (Lee et al., 2019) | Xavier-normal (Glorot & Bengio, 2010) | ImageNet (Russakovsky et al., 2015) |
| CBAM (Woo et al., 2018) | Xavier-uniform (Glorot & Bengio, 2010) | MNIST (LeCun et al., 1998) |
| IEBN (Liang et al., 2019) | Orthogonal (Saxe et al., 2013) | |
| SGENet (Li et al., 2019) | | |
| SEGMENTATION (P.7) | DETECTION (P.7) | BATCH NORMALIZATION (P.8) |
| SegNet (Badrinarayanan et al., 2017) | Faster RCNN (Ren et al., 2015) | VGG |
| PSPNet (Zhao et al., 2017) | | VGG-bn |
| PYTORCH PRETRAIN (P.9) | MATTING (P.7) | LEARNING RATE (P.10) |
| ResNet18/34/50 | Deep image matting (Xu et al., 2017) | Schedule150-225 |
| VGG11/16/19 | AlphaGAN matting (Lutz et al., 2018) | Schedule82-164 |
| STYLE TRANSFER(P.7) | GAN(P.7) | Schedule60-120 |
| Fast neural style (Johnson et al., 2016) | DCGAN (Radford et al., 2015) | Cos-lr (Loshchilov & Hutter, 2016) |

## 3 SIMILARITY

In this section, we further verify the observation that the pruning criteria in Table 2 are highly similar from two perspectives. From an experimental point of view, we use more experiments about image classification to investigate the similarities. From a theoretical perspective, we rigorously show the similarities of the criteria in Table 2 in layer-wise pruning under CWDA.

---

[2]The statistical tests about the situation with or without weight decay can be found in Appendix O.

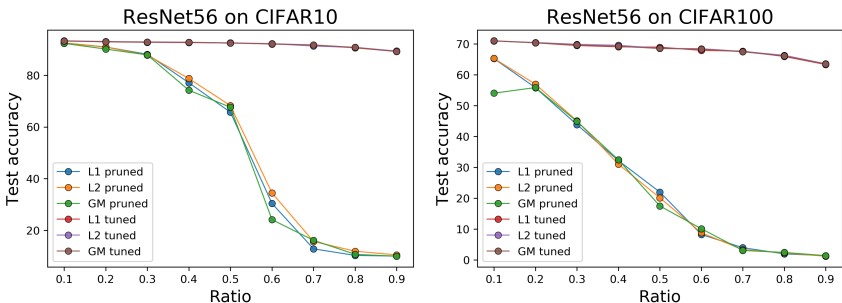

Figure 4: Test accuracy of the ResNet56 on CIFAR10/100 while using different pruning ratios. L1 pruned and L1 tuned denote the test accuracy of the ResNet56 after $\ell_1$ pruning and fine-tuning, respectively. If pruning ratio is equal to 0.5, we prune 50% filters in all layers.

**Empirical Analysis**. (1) In Fig. 4, we show the test accuracy of the ResNet56 after pruning and fine-tuning under using different pruning ratios and datasets. The test accuracy curves of different pruning criteria at different stages are very close under different pruning ratios. This phenomenon implies that those pruned networks using different pruning criteria are very similar, and there are strong similarities among these pruning criteria. The experiments about other commonly used pruning ratio can be found in Appendix N. (2) In Fig. 5, we show the Spearman's rank correlation coefficient (Sp)[3] between different pruning criteria. The Sp in most convolutional layers are more than 0.9, which means the network structures are almost the same after pruning. Note that the Sp in transition layer (*i.e.*, the layer where the dimensions of the filter change, like the layer between stage 1 and stage 2 of ResNet164. The number of dimensions in stage 1 and stage 2 are 16 and 32 respectively.) are relatively small. It is interesting but will not greatly impact the structural similarity of the whole pruned network. The reason for this phenomenon may be that the layers in these areas are sensitive. The similar observations are shown in Fig. 2 in Ding et al. (2019), and Fig. 6 and Fig. 10 in Li et al. (2016).

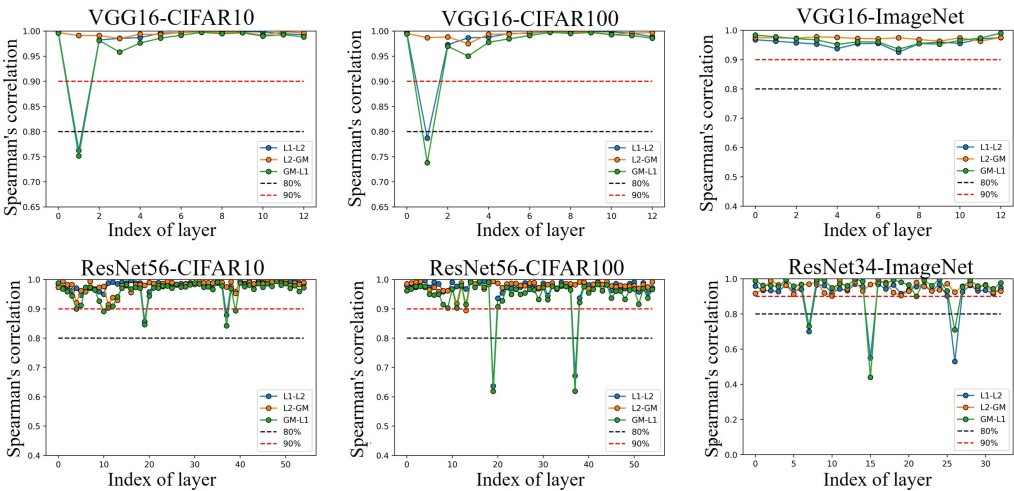

Figure 5: Spearman's rank correlation coefficient (Sp) between different pruning criteria on several networks and datasets (more experiments can be found in Appendix R).

**Theoretical Analysis**. After the verification by experiments, the similarities via using layer-wise pruning among the criteria in Table 2 are proved theoretically in this section. Let $C_1$ and $C_2$ be two pruning criteria to calculate the *importance* for all convolutional filters of one layer. If they

---

[3]Sp is a nonparametric measurement of ranking correlation, and it assesses how well the relationship between two variables can be described using a monotonic function, *i.e.*, filters ranking sequence in the same layer under two criteria in this paper.

can produce the similar ranks of *importance*, we define $C_1$ and $C_2$ are *approximately monotonic* to each other and use $C_1 \cong C_2$ to represent this relationship. In Section 3, we use the Sp to describe this relationship but it's hard to be analyzed theoretically. Therefore, we consider about a stronger condition. Let $\mathbf{X} = (x_1, x_2, ..., x_k)$ and $\mathbf{Y} = (y_1, y_2, ..., y_k)$ be two given sequences. we first normalize their magnitude, *i.e.*, let $\widehat{\mathbf{X}} = \mathbf{X}/\mathbb{E}(\mathbf{X})$ and $\widehat{\mathbf{Y}} = \mathbf{Y}/\mathbb{E}(\mathbf{Y})$ . This operation does not change the ranking sequence of the elements of $\mathbf{X}$ and $\mathbf{Y}$, because $\mathbb{E}(\mathbf{X})$ and $\mathbb{E}(\mathbf{Y})$ are constants, *i.e.*, $\widehat{\mathbf{X}} \cong \widehat{\mathbf{Y}} \Leftrightarrow \mathbf{X} \cong \mathbf{Y}$. After that, if both $\mathbf{Var}(\widehat{\mathbf{X}}/\widehat{\mathbf{Y}})$ and $\mathbf{Var}(\widehat{\mathbf{Y}}/\widehat{\mathbf{X}})$ are small enough, then the Sp between $\mathbf{X}$ and $\mathbf{Y}$ is close to 1, where $\widehat{\mathbf{X}}/\widehat{\mathbf{Y}} = (\widehat{x_1}/\widehat{y_1}, .., \widehat{x_k}/\widehat{y_k})$. The reason is that in these situations, the ratio $\widehat{\mathbf{X}}/\widehat{\mathbf{Y}}$ and $\widehat{\mathbf{Y}}/\widehat{\mathbf{X}}$ will be close to two constants $a, b$. Note that, for any $1 \leq i \leq k$, $\widehat{x_i} \approx a \cdot \widehat{y_i}$ and $\widehat{y_i} \approx b \cdot \widehat{x_i}$. Then, $ab \approx 1$ and $a, b \neq 0$. Therefore, there exists an *approximately monotonic* mapping from $\widehat{y_i}$ to $\widehat{x_i}$ (linear function) and it makes the Sp between $\mathbf{X}$ and $\mathbf{Y}$ close to 1.

**Theorem 1.** *Let* $X \sim N(\mathbf{0}, c^2 \cdot \mathbf{I}_n)$, *and* $(C_1, C_2)$ *is one of* $(\ell_1, \ell_2)$, $(\ell_1, \mathbf{Fermat})$ *or* $(\mathbf{Fermat}, \mathbf{GM})$, *we have*

$$\max\left\{\mathbf{Var}_X\left(\frac{\widehat{C}_2(X)}{\widehat{C}_1(X)}\right), \mathbf{Var}_X\left(\frac{\widehat{C}_1(X)}{\widehat{C}_2(X)}\right)\right\} \lesssim B(n), \tag{3}$$

*where* $\widehat{C}_1(X)$ *denotes* $C_1(X)/\mathbb{E}(C_1(X))$ *and* $\widehat{C}_2(X)$ *denotes* $C_2(X)/\mathbb{E}(C_2(X))$. $B(n)$ *denotes the upper bound of left-hand side and when $n$ is large enough*, $B(n) \to 0$.

*Proof.* (See Appendix G). $\square$

For $i^{\text{th}}$ convolutional layer of a neural network, since $F_{ij}, j = 1, 2, ...N_{i+1}$, meet CWDA and the dimensions of $F_{ij}$ are generally large, we can obtain $\ell_1 \cong \ell_2$, $\ell_2 \cong \mathbf{Fermat}$ and $\mathbf{Fermat} \cong \mathbf{GM}$ according to Theorem 1. Therefore, we have $\ell_1 \cong \ell_2 \cong \mathbf{Fermat} \cong \mathbf{GM}$, which verifies the strong similarities among the criteria shown in Table 2.

## 4 APPLICABILITY

In this section, we analyze the Applicability problem of the Norm-based criteria when we use these criteria to prune a large network (each convolutional layer has a large number of filters). In Fig. 1, we can find that $\ell_2$ can not distinguish the redundancy of Wide ResNet28-10 (regarded as a large network) well from their measured filters' *importance*, because their *importance* are very close (the distribution looks sharp). This problem is related to the variance of *importance*. He et al. (2019) argue that a *small norm deviation* (the values of variance of *importance* are small) makes it difficult to find an appropriate threshold to select filters to prune. However, even if the values of the variance are large, it may still have the Applicability problems. This is because the magnitude of these *importance* may be much greater than the values of the variance, where we can use the mean of *importance* to represent their magnitude. We believe that the following two situations may cause Applicability problem: for the filters $F_i$ in $i^{\text{th}}$ convolutional layer,

**(1)** If the mean $\mathbb{E}(F_i) = 0$ and the variance $\mathbf{Var}(F_i)$ is close to 0;

**(2)** If $\mathbb{E}(F_i) \neq 0$ and $\mathbf{Var}(F_i)/\mathbb{E}(F_i)$ is close to 0.

For a network with large number of convolutional filters, it's easy to know that the dimensions of the filters are also large enough. For $\ell_2$ pruning, according to CWDA (the proof in Appendix L), we can obtain that the mean of $\ell_2(F_i)$ is $\sqrt{2}\sigma_i \cdot \Gamma(\frac{k_i+1}{2})/\Gamma(\frac{k_i}{2}) \neq 0$, where $\sigma_i$ and $k_i$ are the standard deviation and dimension of the parameters in $i^{\text{th}}$ layer, respectively. And $\mathbf{Var}(F_i)/\mathbb{E}(F_i) \to 0$ when $k_i$ is large enough. From these reasons, the *importance* measured by $\ell_2$ norm tends to be identical, *i.e.*, it's hard to distinguish the network redundancy well in this situation. Moreover, from the proof in Appendix H, we know that the Fermat point $\mathbf{F}$ of $F_i$ and the origin $\mathbf{0}$ approximately coincide. According to Table 2, $||\mathbf{F} - F_i||_2 \approx ||\mathbf{0} - F_i||_2 = ||F_i||_2$. Therefore, the *importance* of $\mathbf{Fermat}$ and $\ell_2$ are almost equivalent when CWDA holds. Hence, a similar conclusion can be obtained for $\mathbf{Fermat}$ criterion. Intuitively, the $\ell_1$ criterion should have the same Applicability problem as the $\ell_2$ criterion. However, from the proof in Appendix L, the mean of $\ell_1(F_i)$ is $\sigma_i \cdot k\sqrt{\frac{2}{\pi}} \neq 0$, but

$\mathbf{Var}(F_i)/\mathbb{E}(F_i)$ tends to be a none-zero constant with respect to $\sigma_i$. In other words, $\ell_1$ criterion does not necessarily have Applicability problems unless $\sigma_i$ is small enough.

Except for the Norm-based criteria, we analyze another type pf pruning criteria called RePr (Prakash et al., 2019). This criterion considers the orthogonality among the filters in one layer. Based on the proof in Appendix L and the fact that the *importance* measured by $\ell_2$ norm tend to be identical, we also know that this criterion cannot prune the network well when the number of filters is too large. Moreover, in Section 5.2, we study the Applicability problem for more different types of pruning criteria from a numerical perspective.

## 5 DISCUSSION

### 5.1 WHY CWDA SOMETIMES DOES NOT HOLD?

CWDA may not always hold. As shown in Appendix P, a small number of convolutional filters may not pass the statistical tests. In this section, we try to analyze this phenomenon.

(1) **Need to be trained well enough**. The distribution of parameters can only be discussed when the network is trained well. If the network does not converge, it is easy to construct a scenario which does not satisfy CWDA, *e.g.*, in Fig. 16 (Appendix D), we train a network with uniform initialization. Although the distribution of parameters converges with the increase of epoch more and more to a normal distribution, when only a few epochs are trained, the distribution of parameters is closer to a uniform distribution. At this time, the distribution obviously does not satisfy CWDA. Moreover, if a network is not trained well, *e.g.*, its parameters converge to bad local minima, there may be many unforeseen circumstances causing CWDA to be unsatisfied. In order to eliminate these factors, the network should be trained when studying the parameter distribution.

(2) **The number of filters is insufficient.** In Appendix P, the layers that can not pass the statistical tests are almost those whose position is in the front of the network. A common feature of these layers is that they have a few filters, which may not estimate statistics well. In Fig. 5, due to the sensibility of layers, the Sp in the transition layer are relatively small. Taking the second convolutional layer (64 filters) in VGG16 on CIFAR10 as an example, we find that increasing the number of filters could alleviate this sensitivity. As shown in Fig. 6, we change the number of filters in this layer from 64 to 128 or 256. After that, the Sp increases significantly, and it suggests that enough number of filters are essential.

(3) **The dimensions of the filter are not large enough.** The dimension of a filter in the $i^{\text{th}}$ layer is closely related to the number of filters in the $(i-1)^{\text{th}}$ layer. To eliminate the influence from the number of the filters, we can consider the correlation matrix of each parameter in the filter. As the analysis in Section 2, only when the dimension of the convolutional filter is large enough will the matrix approximate a diagonal matrix. It may also be a reason why the layer in the front of the network usually can not pass the statistical test in Appendix P.

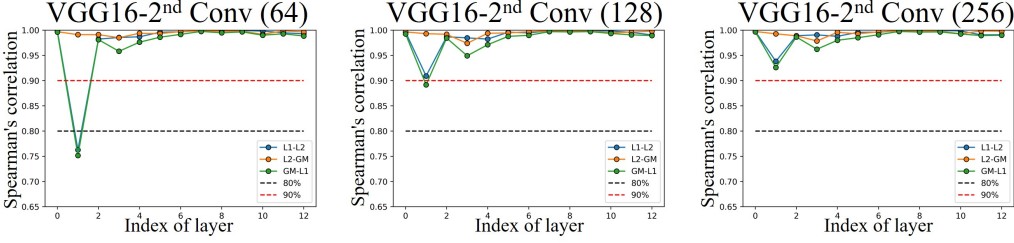

Figure 6: The Sp between different pruning criteria on VGG16 (CIFAR10). The number of filters in the second convolutional layers is changed from 64 to 256.

### 5.2 WHAT ABOUT OTHER PRUNING CRITERIA?

In this section, we study the Similarity and Applicability problem in more types of pruning criteria through numerical experiments, such as Activation-based pruning (Hu et al., 2016; Luo & Wu,

2017), Importance-based pruning (Molchanov et al., 2016; 2019a) and BN-based pruning (Liu et al., 2017b).

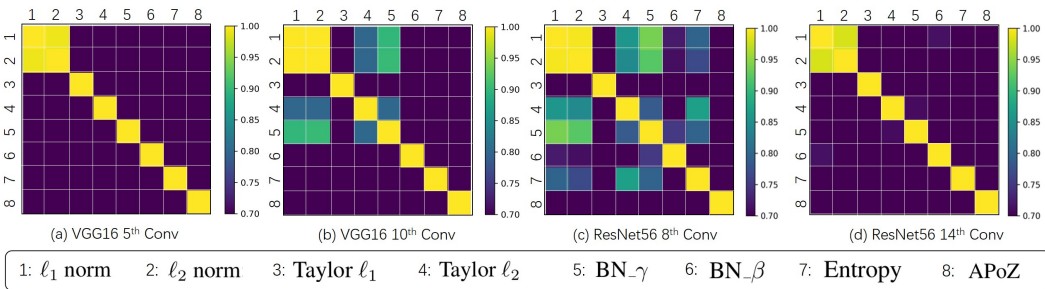

(a) VGG16 5$^{th}$ Conv    (b) VGG16 10$^{th}$ Conv    (c) ResNet56 8$^{th}$ Conv    (d) ResNet56 14$^{th}$ Conv

1: $\ell_1$ norm    2: $\ell_2$ norm    3: Taylor $\ell_1$    4: Taylor $\ell_2$    5: BN_$\gamma$    6: BN_$\beta$    7: Entropy    8: APoZ

Figure 7: The Sp between different types of pruning criteria on VGG16 and ResNet56.

For each type, we choose two representative criteria and we call them: (1) Norm-based: $\ell_1$ and $\ell_2$; (2) Importance-based: Taylor $\ell_1$ and Taylor $\ell_2$ (Molchanov et al., 2016; 2019a;b); (3) BN-based: BN_$\gamma$[4] and BN_$\beta$ (Liu et al., 2017b); (4) Activation-based: Entropy (Luo & Wu, 2017) and APoZ (Hu et al., 2016). The details of these criteria can be found in Appendix M.

**The Similarity for different types of pruning criteria**. In Fig. 7, we show the Sp between different types of pruning criteria, and only the Sp greater than 0.7 are shown because if Sp < 0.7, it means that there is no strong similarity between two criteria in the current layer. According to the Sp shown in Fig. 7, we can have the following observations: (1) As verified in Section 3, $\ell_1$ and $\ell_2$ maintain a strong similarity in each layer; (2) In the layers shown in Fig. 7(a) and Fig. 7(d), the Sp between most different pruning criteria are not large in these layers, which indicates that these methods have great differences in the redundancy measurement of convolution filters. This may lead to a phenomenon that one criterion considers a convolutional filter to be important, while another considers it redundant. We find a specific example which is shown in Appendix E; (3) Intuitively, the same type of criteria should be similar. However, it can be seen from Fig. 7(b) and Fig. 7(c) that the Sp between Taylor $\ell_1$ and Taylor $\ell_2$ is not large, but Taylor $\ell_2$ has strong similarity with both two Norm-based criteria. Moreover, the Sp between BN_$\gamma$ and each Norm-based criteria exceeds 0.9, but it is not large in other layers (Fig. 7(b) and Fig. 7(d)). These phenomena are worthy of further study.

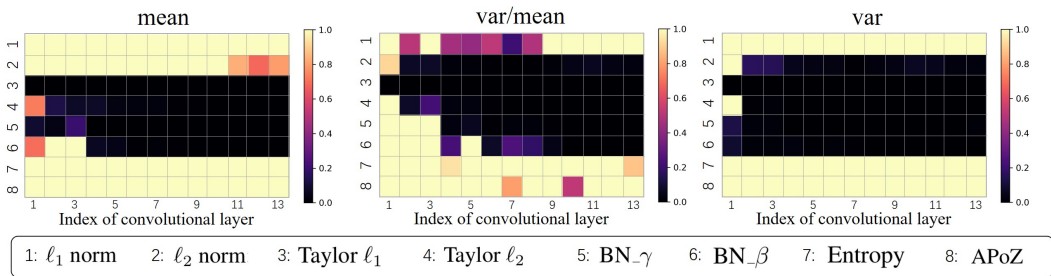

1: $\ell_1$ norm    2: $\ell_2$ norm    3: Taylor $\ell_1$    4: Taylor $\ell_2$    5: BN_$\gamma$    6: BN_$\beta$    7: Entropy    8: APoZ

Figure 8: The visualization of Applicability problem for different types of criteria. (VGG16)

**The Applicability for different types of pruning criteria**. According to the analysis in Section 4, the Applicability problem depends on the mean and variance of the parameters. Fig. 8 shows the result of the *importance* measured by different pruning criteria on each layer of VGG16. VGG16 is a network whose width becomes larger gradually as the depth increases (from 64 to 512). For this reason, we can study the Applicability problem in generally wide (shallow layers) and sufficiently wide (deep layers) convolutional layers. First, we analyze the Norm-based criteria. The mean of both $\ell_1$ and $\ell_2$ are relatively large, but in most layers, the variance/mean of $\ell_2$ is much smaller than that of $\ell_1$, which means that the $\ell_2$ pruning has Applicability problem, while the $\ell_1$ does not. This is consistent with the conclusion in Section 4. Next, for the Activation-based criteria, the

---

[4]The empirical result for slimming training (Liu et al., 2017b) is shown in Appendix B.

mean and variance/mean are both large, which means that these two Activation-based criteria can distinguish the network redundancy well from their measured filters' *importance*. However, for the Importance-based and BN-based criteria, their mean and variance/mean are close to 0. According to the condition (1) shown in Section 4, these criteria have Applicability problem, especially in the deep layers.

## 5.3 WHAT ABOUT GLOBAL PRUNING?

Compared with layer-wise pruning, global pruning is more widely (Liu et al., 2018; Molchanov et al., 2016; Liu et al., 2017b) used in channel pruning.

**Similarity while using global pruning**. In Fig. 9, we show the similarity of different types of pruning criteria using global pruning on VGG16 and ResNet56. Comparing to the results from the layer-wise pruning shown in Fig. 7, we can find that the similarities of most pruning criteria are quite different in global pruning. In particular, for the results of $\ell_1$ and $\ell_2$ in Fig. 9(a), the similarity between $\ell_1$ and $\ell_2$ is not as strong as the one in the layer-wise case. We argue that this phenomenon may be due to the differences between the parameter distribution in different convolutional layers. More analysis can be found in Appendix A.

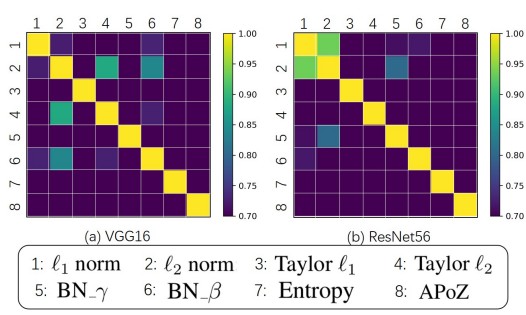

|    |    |    |    |
|----|----|----|----|
| 1: $\ell_1$ norm | 2: $\ell_2$ norm | 3: Taylor $\ell_1$ | 4: Taylor $\ell_2$ |
| 5: BN_$\gamma$ | 6: BN_$\beta$ | 7: Entropy | 8: APoZ |

Figure 9: Similarity while using global pruning.

**Applicability while using global pruning**. In fact, for global pruning, Norm-based criteria are not prone to Applicability problems. From Section 4, we have the estimations for the magnitude of *importance* in $i^{\text{th}}$ layer calculated by $\ell_1$ and $\ell_2$ as $\sigma_i \cdot k \sqrt{\frac{2}{\pi}}$ and $\sqrt{2}\sigma_i \cdot \Gamma(\frac{k_i+1}{2})/\Gamma(\frac{k_i}{2})$. Since $\sigma_i$ and $k_i$ are quite different, the variance of the *importance* is large in this situation. Fig. 10 shows this kind of difference of the magnitude on different convolutional layers.

Our estimation also explains a common problem in practical applications of global pruning: the network is easily pruned off. As shown in Fig. 10, we take ResNet56 as an example. Since the *importance* in first stage is much smaller than the *importance* in the deeper layer, global pruning will give priority to prune the convolutional filters of the first stage. To solve the problem of inconsistent magnitude, we suggest that some normalization methods should be implemented or a protection mechanism should be established, *e.g.*, a mechanism which can ensure that each layer has at least a certain number of convolutional filters that will not be pruned.

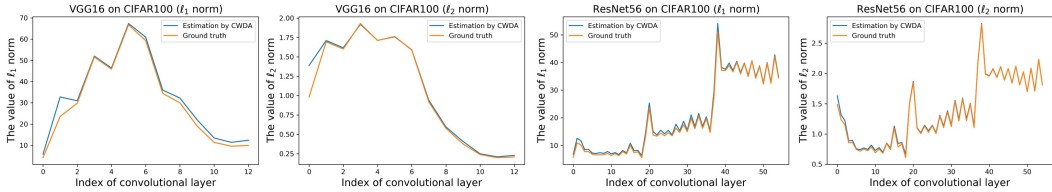

Figure 10: The magnitude of the *importance* measured by $\ell_1$ and $\ell_2$ criteria. The actual magnitude almost coincides with the estimation obtained by CWDA.

## 6 CONCLUSION

In this paper, we found two blind spots on pruning criteria: Similarity and Applicability. For Similarity, some primary pruning criteria can obtain very similar pruning results. For Applicability, some criteria can not distinguish the redundancy of the filters well when the number of filters is large enough. The comprehensive experiments validate these two findings, and our assumption is called CWDA. Under CWDA, these two blind spots are also discussed when using global pruning.

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

# A  ADDITIONAL STUDY FOR CWDA

## A.1  IS NORMAL DISTRIBUTION NECESSARY?

In this section, we use the random functions provided by NumPy[5] to generate data manually to study the influence of distribution on the similarity between Norm-based criteria. We use these distribution functions to generate 100 vectors with 50 dimensions each time. Each vector can be regarded as a convolutional filter with dimension 50.

As shown in the Table 4 (Left), we compare the Sp of these generated convolutional filters between $\ell_1$, $\ell_2$ and **GM** criteria. If Sp is greater than 0.9, we mark it as green. Otherwise, we mark it as red. Under these generated distribution, $\ell_1$ and $\ell_2$ can keep larger Sp in most cases, which indicates that they still maintain strong similarity for these distribution. But the similarity between **GM** and $\ell_1$ (or $\ell_2$) is relatively poor.

Table 4: The similarity when using different distribution.

| Distribution | Mean | Std | $\ell_1$-$\ell_2$ | **GM**-$\ell_2$ | **GM**-$\ell_1$ | Mean | Std | $\ell_1$-$\ell_2$ | **GM**-$\ell_2$ | **GM**-$\ell_1$ |
|---|---|---|---|---|---|---|---|---|---|---|
| rand | 0.50 | 0.29 | 0.94 | 0.31 | 0.03 | **0.00** | 0.29 | 0.96 | 0.97 | 0.95 |
| randn | 0.00 | 1.00 | 0.93 | 0.95 | 0.93 | **0.00** | 0.99 | 0.95 | 0.93 | 0.94 |
| 1+randn | 1.00 | 1.00 | 0.95 | 0.63 | 0.51 | **0.00** | 1.00 | 0.95 | 0.96 | 0.95 |
| beta(1,2) | 0.33 | 0.24 | 0.94 | 0.44 | 0.21 | **0.00** | 0.23 | 0.96 | 0.97 | 0.96 |
| beta(0.1,2) | 0.04 | 0.11 | 0.92 | 0.96 | 0.95 | **0.00** | 0.12 | 0.95 | 0.97 | 0.98 |
| beta(2,5) | 0.28 | 0.15 | 0.96 | 0.70 | 0.49 | **0.00** | 0.15 | 0.91 | 0.94 | 0.95 |
| chisquare(2) | 2.02 | 2.03 | 0.92 | 0.91 | 0.73 | **0.00** | 2.01 | 0.97 | 0.95 | 0.94 |
| gamma(2,0.5) | 1.01 | 0.71 | 0.92 | 0.79 | 0.55 | **0.00** | 0.71 | 0.92 | 0.96 | 0.97 |
| gamma(2,2) | 3.93 | 2.76 | 0.94 | 0.86 | 0.69 | **0.00** | 2.73 | 0.91 | 0.96 | 0.96 |
| beta(1,2)+beta(0.1,2) | 0.37 | 0.26 | 0.81 | 0.67 | 0.31 | **0.00** | 0.26 | 0.81 | 0.91 | 0.96 |
| beta(1,2)+beta(2,5) | 0.61 | 0.27 | 0.93 | 0.44 | 0.17 | **0.00** | 0.27 | 0.93 | 0.94 | 0.94 |

If we use zero-mean normalization for the generated data $X$, *i.e.*, $X - \mathbb{E}X$, as shown in Table4 (Right), we also find that the Sp between these three criteria are greater than 0.9 for all distribution. This phenomenon may indicate that the normality in CWDA is not necessary for the similarity of criteria, but the zero mean is the key to the similarity.

Table 5: The similarity when using *mixed* distribution.

| Distribution 1 | Distribution 2 | Use zero-mean? | Mean | Std | $\ell_1$-$\ell_2$ | **GM**-$\ell_2$ | **GM**-$\ell_1$ |
|---|---|---|---|---|---|---|---|
| randn | randn×8 | | 0.00 | 5.69 | 0.98 | 0.95 | 0.96 |
| randn | rand | | 0.26 | 0.78 | 0.97 | 0.96 | 0.96 |
| randn | rand | ✓ | 0.00 | 0.78 | 0.99 | 0.98 | 0.98 |
| randn | rand+5 | | 2.72 | 2.87 | 0.99 | -0.28 | -0.24 |
| randn | rand+5 | ✓ | 0.00 | 2.85 | 0.71 | 0.91 | 0.88 |
| randn | log(rand) | | -0.49 | 1.08 | 0.85 | 0.14 | 0.49 |
| randn | log(rand) | ✓ | 0.00 | 1.07 | 0.69 | 0.90 | 0.89 |
| rand | log(rand) | | -0.26 | 1.08 | 0.98 | 0.87 | 0.87 |
| rand | log(rand) | ✓ | 0.00 | 1.01 | 0.65 | 0.96 | 0.81 |

Next, we consider the *mixed* distribution, *i.e.*, as shown in Table 5, among the 100 vectors we generated, the first 50 vectors are generated by Distribution 1, and the last 50 vectors are generated by Distribution 2. There are several notable phenomena in Table 5:

(1) For the *mixed* distribution composed of two normal distribution with zero-mean but different variances, the three criteria can maintain strong similarity;

(2) In the *mixed* distribution, zero-mean normalization may not make the Sp between the three criteria stronger, which is inconsistent with the situation in Table 4.

(3) For $\ell_1$ and $\ell_2$, the Sp measured by the *mixed* distribution is lower than single distribution (Table 4) in general.

According to CWDA, the *mixed* distribution in phenomenon (1) actually corresponds to global pruning strategy. If CWDA holds, the distribution of parameters in different convolutional layers depends

---

[5]https://docs.scipy.org/doc/numpy-1.13.0/reference/routines.random.html

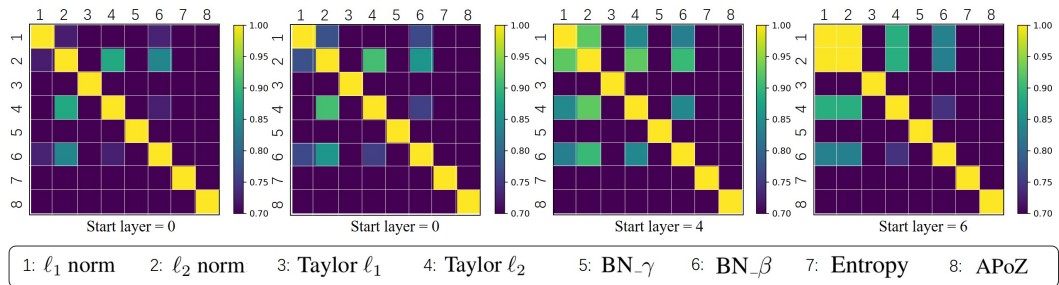

Figure 11: The similarity for different type of pruning criteria while using global pruning with different start layer.

on the variance of parameters in each layer, and these variances are generally different. Therefore, the global pruning strategy is actually to prune the convolutional filters that satisfy different Gaussian distribution. Unlike layer-wise pruning, these filters cannot be represented by a general random variable $F$, where $F \sim \mathbf{N}(\mathbf{0}, c^2 \cdot \mathbf{I}_{N_i \times k \times k})$. It needs to be described by a stochastic process, *i.e.*, $F_t \sim \mathbf{N}(\mathbf{0}, c^2 \cdot \mathbf{I}_{N_i \times k \times k}^{(t)})$.

According to the experiment of phenomenon (1) and CWDA, we expect that $\ell_1$ and $\ell_2$ should still maintain strong similarity while using the global pruning. However, as shown in Fig. 9, the Sp of $\ell_1$ and $\ell_2$ is not very large. This seems to contradict the experiment of phenomenon (1). In fact, however, according to the analysis in Section 9, CWDA may not hold in the first few layers of CNNs. So if these convolutional filters are also considered to be globally pruned in global pruning, then the similarity between $\ell_1$ and $\ell_2$ may be weakened, which is consistent with the experiment in Table 5.

To verify this statement, Fig. 11 shows the result of VGG16 via using global pruning strategy. The *Start layer* means the layer that we start to use for global pruning. For example, when *Start layer* is equal to 0, it is the general global pruning; When *Start layer* is equal to 5, the filters from the first five layers are not be considered to be pruned. In Fig. 11, we can see that, with the increase of *start layer*, the similarity between $\ell_1$ and $\ell_2$ becomes stronger and stronger. This shows that the similarity between $\ell_1$ and $\ell_2$ is not as strong as layer-wise in global pruning, which is mainly caused by the first few layers that do not satisfy CWDA.

Moreover, in phenomenon (2), global pruning is different from layer-wise pruning shown in Table 4. The zero-mean normalization may not make these pruning criteria similar. Therefore, global pruning is worthy of further studies, such as using the stochastic process to explore the similarity and the Applicability problem of different types of pruning criteria.

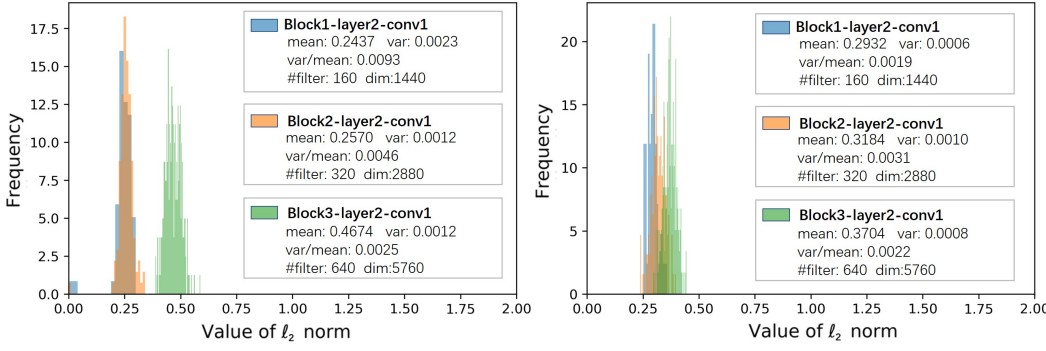

Figure 12: The distribution of $\ell_2$ norm when the WRN is not trained well. Left: without data augmentation. Right: trained with uniform initialization and 6 epochs. Like the phenomenon in Fig. 1, $\ell_2$ pruning still has Applicability problems when the network is not trained well.

# B  TRAINING THROUGH SLIMMING

Figure 13: The Similarity for different criteria with/without slimming (Liu et al., 2017a).

As a representative of the BN-based pruning method, slimming pruning(Liu et al., 2017a) can not be directly compared with the criteria mentioned in the paper because it adopts a special training method. Therefore, we use the training method in Liu et al. (2017a) to train another ResNet56 on cifar100. Then, the analysis of similarities between 8 different pruning criteria on such a model is shown in Fig. 13.

In this situation, the fifth criterion $BN_{-\gamma}$ is the method introduced in Liu et al. (2017a). From Fig. 13, there is no significant difference in the result of the similarity between ResNet56 obtained by slimming method and resnet56 trained in general.

## C  THE RESULT OF SP

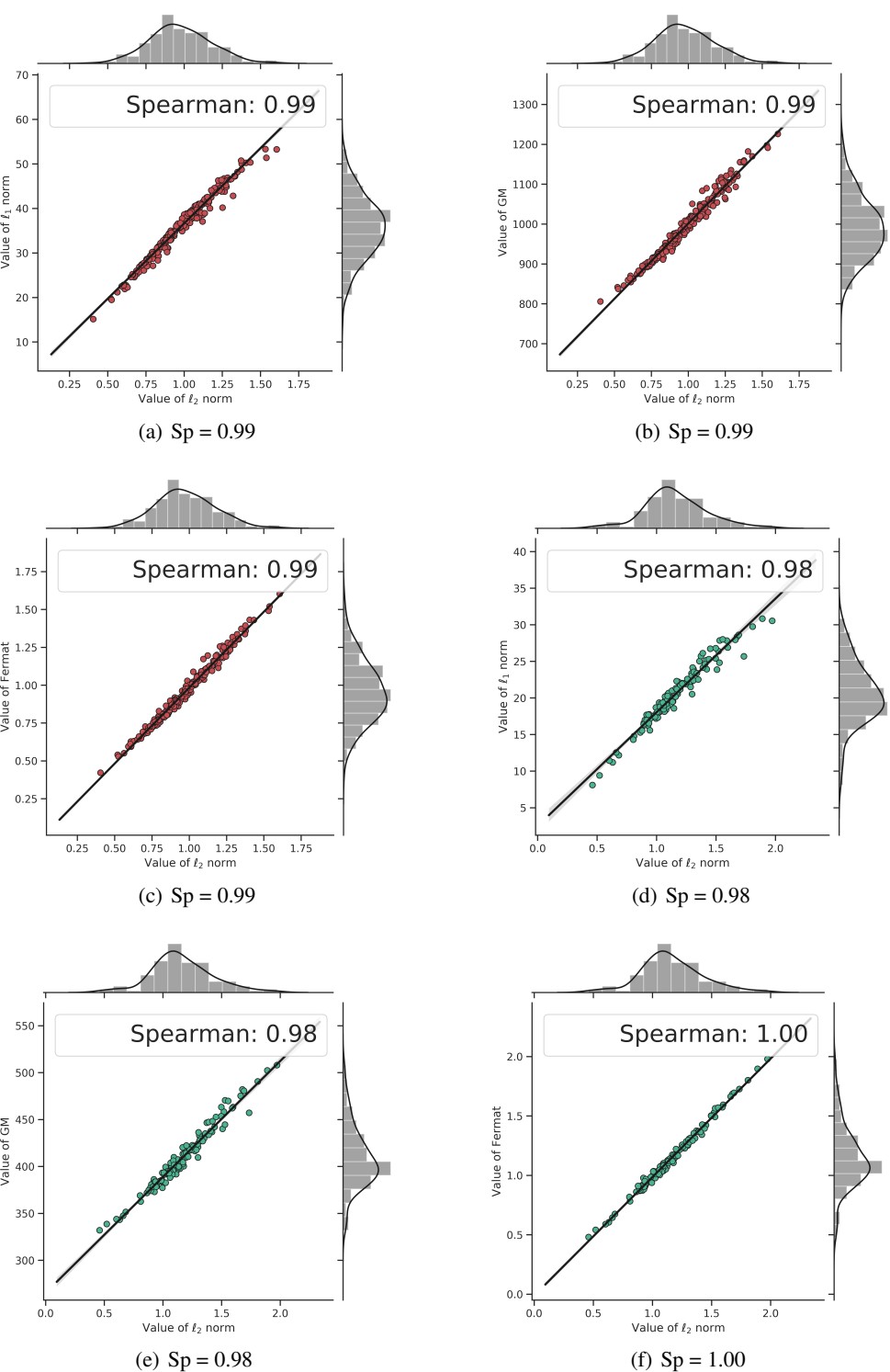

(a) Sp = 0.99

(b) Sp = 0.99

(c) Sp = 0.99

(d) Sp = 0.98

(e) Sp = 0.98

(f) Sp = 1.00

Figure 14: The Spearman's rank correlation coefficient (Sp) for different criteria. (a-c) are Sp between $\ell_1$ and $\ell_2$, **GM** and $\ell_2$, **Fermat** and $\ell_2$ from ResNet18 ($12^{\text{th}}$ Conv), respectively. The results of VGG16 ($3^{\text{rd}}$ Conv) are shown in (d-f). If the Sp of two pruning criteria is close to 1, then the sequence of their pruned filters may have strong similarity.

# D    OTHER RESULT

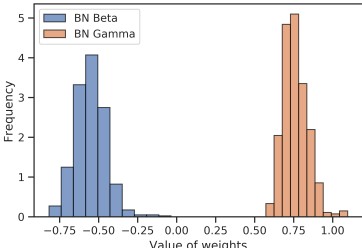 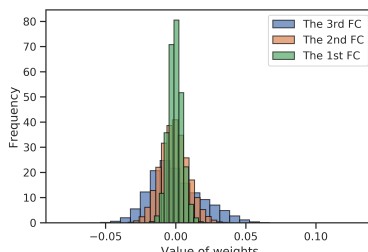

Figure 15: The distribution about other learnable parameters. (Left): The disrtibution about the learnable parameters of batch normalization. (Rihgt): The distribution of the parameters of fully-connected layers (FC). For FC, the Sp between the criteria in Table2 are greater than 0.9. More analysis can be found in Appendix A.

In Fig 15, we show the other learnable parameters (*i.e.* Batch normalization (BN) and fully-connected neural network (FC)) in VGG16-BN. For BN, the distribution of its parameters does not satisfy CWDA, and similar results are shown in Liu et al. (2017a); Tian et al. (2019). Moreover, the learnable parameters of fully-connected layers also do not follow a Gaussian-alike distribution, which is consistent with the conclusion in previous work Bellido & Fiesler (1993); Neal (1995); Go et al. (2004).

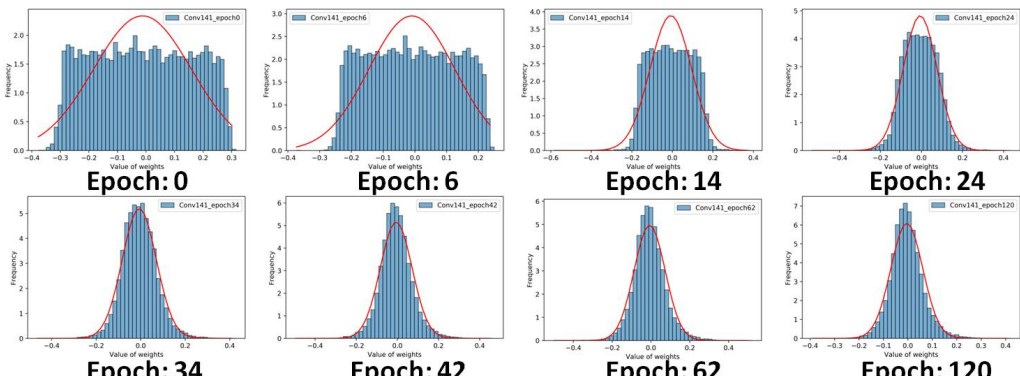

Figure 16: The distribution of the convolutional filter (141[th] Conv) with kaiming-uniform initialization for each epoch.

# E  AN INTERESTING CASE FOR *importance* MEASURED BY DIFFERENT CRITERIA

The following results are the index of pruned filters obtained by the filters' *importance* from different types of pruning criteria. We take VGG16 ($2^{nd}$) as an example. The $5^{th}$ filter in this layer is regarded as a redundant convolutional filter for APoZ criterion, but other criteria consider it to be almost the most important.

Taylor $\ell_1$: [27, 36, 25, 11, 6, 23, 24, 16, 0, 57, 48, 53, 1, 61, 18, 55, 34, 15, 51, 58, 31, 3, 12, 21, 59, 30, 7, 38, 41, 50, 10, 33, 17, 46, 62, 13, 49, 43, 42, 47, 2, 32, 44, 20, 39, 52, 56, 40, 9, 26, 37, 22, 29, 54, 60, 8, 14, 45, 4, 63, 19, 35, 28, **5**]

Taylor $\ell_2$: [23, 32, 36, 11, 62, 16, 30, 59, 10, 13, 2, 50, 38, 0, 46, 43, 21, 26, 15, 22, 7, 51, 39, 33, 14, 58, 9, 40, 57, 6, 61, 44, 20, 48, 3, 53, 41, 56, 17, 12, 18, 31, 4, 1, 25, 19, 63, 24, 54, 45, 52, 37, 55, 47, 34, 35, 8, 29, 42, 27, 49, 28, 60, **5**]

BN_$\beta$: [52, 46, 32, 21, 14, 29, 17, 0, 19, 36, 1, 51, 44, 40, 41, 60, 57, 27, 22, 53, 63, 8, 30, 26, 23, 58, 39, 18, 9, 47, 31, 35, 11, 37, 55, 45, 3, 61, 6, 4, 33, 25, 15, 48, 43, 28, 56, 2, 13, 16, 34, 20, 59, 10, 7, 24, 50, 62, 12, 49, 38, 42, **5**, 54]

APoZ: [**5**, 10, 38, 42, 62, 24, 13, 12, 7, 28, 59, 15, 23, 11, 16, 56, 34, 35, 57, 19, 2, 49, 43, 25, 6, 63, 61, 36, 9, 27, 33, 20, 48, 58, 55, 18, 51, 31, 1, 0, 53, 37, 26, 29, 47, 60, 8, 44, 41, 46, 21, 17, 14, 32, 52, 22, 39, 3, 40, 30, 4, 45, 50, 54]

## F    RELATED PROPOSITION

**Proposition 1** (Amoroso distribution). *The Amoroso distribution is a four parameter, continuous, univariate, unimodal probability density, with semi-infinite range (Crooks, 2012). And its probability density function is*

$$\mathbf{Amoroso}(X|a,\theta,\alpha,\beta) = \frac{1}{\Gamma(\alpha)}|\frac{\beta}{\theta}|(\frac{X-a}{\theta})^{\alpha\beta-1}\exp\left\{-(\frac{X-a}{\theta})^{\beta}\right\}, \tag{4}$$

*for $x,a,\theta,\alpha,\beta \in \mathbb{R}, \alpha > 0$ and range $x \geq a$ if $\theta > 0$, $x \leq a$ if $\theta < 0$. The mean and variance of Amoroso distribution are*

$$\mathbb{E}_{X\sim\mathbf{Amoroso}(X|a,\theta,\alpha,\beta)}X = a + \theta \cdot \frac{\Gamma(\alpha+\frac{1}{\beta})}{\Gamma(\alpha)}, \tag{5}$$

*and*

$$\mathbf{Var}_{X\sim\mathbf{Amoroso}(X|a,\theta,\alpha,\beta)}X = \theta^2\left[\frac{\Gamma(\alpha+\frac{2}{\beta})}{\Gamma(\alpha)} - \frac{\Gamma(\alpha+\frac{1}{\beta})^2}{\Gamma(\alpha)^2}\right]. \tag{6}$$

**Proposition 2** (Half-normal distribution). *Let random variable $X$ follow a normal distribution $N(0,\sigma^2)$, then $Y = |X|$ follows a half-normal distribution (Pescim et al., 2010). Moreover, $Y$ also follows $\mathbf{Amoroso}(x|0,\sqrt{2}\sigma,\frac{1}{2},2)$. By Eq. (5) and Eq. (6), the mean and variance of half-normal distribution are*

$$\mathbb{E}_{X\sim N(0,\sigma^2)}|X| = \sigma\sqrt{2/\pi}, \tag{7}$$

*and*

$$\mathbf{Var}_{X\sim N(0,\sigma^2)}|X| = \sigma^2\left(1 - \frac{2}{\pi}\right). \tag{8}$$

**Proposition 3** (Scaled Chi distribution). *Let $X = (x_1,x_2,...x_k)$ and $x_i, i = 1,...,k$ are $k$ independent, normally distributed random variables with mean 0 and standard deviation $\sigma$. The statistic $\ell_2(X) = \sqrt{\sum_{i=1}^{k} x_i^2}$ follows Scaled Chi distribution (Crooks, 2012). Moreover, $\ell_2(X)$ also follows $\mathbf{Amoroso}(x|0,\sqrt{2}\sigma,\frac{k}{2},2)$. By Eq. (5) and Eq. (6), the mean and variance of Scaled Chi distribution are*

$$\mathbb{E}_{X\sim N(\mathbf{0},\sigma^2\cdot\mathbf{I_k})}[\ell_2(X)]^j = 2^{j/2}\sigma^j \cdot \frac{\Gamma(\frac{k+j}{2})}{\Gamma(\frac{k}{2})}, \tag{9}$$

*and*

$$\mathbf{Var}_{X\sim N(\mathbf{0},\sigma^2\cdot\mathbf{I_k})}\ell_2(X) = 2\sigma^2\left[\frac{\Gamma(\frac{k}{2}+1)}{\Gamma(\frac{k}{2})} - \frac{\Gamma(\frac{k+1}{2})^2}{\Gamma(\frac{k}{2})^2}\right]. \tag{10}$$

## G    PROOF OF THEOREM 1

**Theorem 1.** Let $X \sim N(\mathbf{0},c^2\cdot\mathbf{I}_n)$ and $(C_1,C_2)$ is one of $(\ell_1,\ell_2)$, $(\ell_1,\mathbf{Fermat})$ or $(\mathbf{Fermat},\mathbf{GM})$, we have

$$\max\left\{\mathbf{Var}_X\left(\frac{\widehat{C}_2(X)}{\widehat{C}_1(X)}\right), \mathbf{Var}_X\left(\frac{\widehat{C}_1(X)}{\widehat{C}_2(X)}\right)\right\} \lesssim B(n). \tag{11}$$

where $\widehat{C}_1(X)$ denotes $C_1(X)/\mathbb{E}(C_1(X))$ and $\widehat{C}_2(X)$ denotes $C_2(X)/\mathbb{E}(C_2(X))$. $B(n)$ denotes the upper bound of left-hand side and when $n$ is large enough, $B(n) \to 0$.

For $i_{th}$ layer, we use $v_j$ to represent $F_{ij}, j = 1,2,...N$. And $v_j$ meets CWDA (*i.e.*, $v_j$ are i.i.d and $v_j \sim N(0,c^2\cdot\mathbf{I})$). Actually, from the theoretical analysis in Section 3, the fact that two criteria $C_1$ and $C_2$ meet Eq.11 is equivalent to $C_1 \cong C_2$.

**(1) For** $(\ell_2, \ell_1)$. In fact, $\ell_2 \cong \ell_1$ (their importance rankings are similar) is not trivial. Generally speaking, for convolutional filters, $\mathbf{dim}(v_j)$ is large enough. Since $v_i$ satisfies CWDA, from Theorem 2, we know that the variance of ratio between $\widehat{\ell}_1$ and $\widehat{\ell}_2$ have a bound $O(\mathbf{dim}(v_j)^{-1})$, which means $\ell_2$ and $\ell_1$ are *appropriate monotonic*. Specific numerical validation is shown in Fig. 17 of Appendix H).

**Theorem 2.** *Let* $X \sim N(\mathbf{0}, c^2 \cdot \mathbf{I}_n)$, *we have*

$$\mathbf{max}\left\{\mathbf{Var}_X\left(\frac{\widehat{\ell}_2(X)}{\widehat{\ell}_1(X)}\right), \mathbf{Var}_X\left(\frac{\widehat{\ell}_1(X)}{\widehat{\ell}_2(X)}\right)\right\} \lesssim \frac{1}{n}. \tag{12}$$

*where* $\widehat{\ell}_1(X)$ *denotes* $\ell_1(X)/\mathbb{E}(\ell_1(X))$ *and* $\widehat{\ell}_2(X)$ *denotes* $\ell_2(X)/\mathbb{E}(\ell_2(X))$.

*Proof.* (See Appendix H). $\qquad\square$

**(2) For** $(\ell_1, \mathbf{Fermat})$. Since $v_i$ satisfies CWDA, from Theorem 3, we know that the Fermat point of $v_i$ and the origin $\mathbf{0}$ approximately coincide. According to Table 2, $||\mathbf{Fermat} - v_i||_2 \approx ||\mathbf{0} - v_i||_2 = ||v_i||_2$. Therefore, from Theorem 2, the bound $B(n)$ for the $(\ell_1, \mathbf{Fermat})$ is also $\frac{1}{n}$. Moreover, since CWDA, the centroid of $v_i$ is $\mathbf{G} = \frac{1}{n}\sum_{i=1}^{N} v_i = \mathbf{0}$. Hence,

$$\mathbf{G} = \mathbf{0} \approx \mathbf{Fermat}. \tag{13}$$

**Theorem 3.** *Let random variable* $v_i \in \mathbb{R}^k$ *and they are i.i.d and follow normal distribution* $N(\mathbf{0}, \sigma\mathbf{I}_k)$. *For* $F \in \mathbb{R}^k$, *we have* $\mathbf{argmin}_F\left\{\mathbb{E}_{v_i \sim N(\mathbf{0},\sigma\mathbf{I}_k)}\sum_{i=1}^{n}||F - v_i||_2\right\} = \mathbf{0}$.

*Proof.* (See Appendix I). $\qquad\square$

**(3) For** $(\mathbf{GM}, \mathbf{Fermat})$. First, we show the following two theorems:

**Theorem 4.** *For* $n$ *random variables* $a_i \in \mathbb{R}^k$ *follow* $N(\mathbf{0}, c^2 \cdot \mathbf{I}_k)$.*When* $k$ *is large enough, we have such an estimation:*

$$\mathbf{Var}_{a_i}\frac{F_1(a_i)}{F_2(a_i)} \approx \frac{1}{2nk}, \quad \mathbf{Var}_{a_i}\frac{F_2(a_i)}{F_1(a_i)} \approx \frac{1}{2nk}, \tag{14}$$

*where* $F_1(a_i) = \sum_{i=1}^{n}||a_i||_2/\mathbb{E}(\sum_{i=1}^{n}||a_i||_2)$ *and* $F_2(a_i) = \sum_{i=1}^{n}||a_i||_2^2/\mathbb{E}(\sum_{i=1}^{n}||a_i||_2^2)$.

*Proof.* (See Appendix J). $\qquad\square$

**Theorem 5.** *Let* $v_0, v_1, ..., v_k$ *be the* $k + 1$ *vectors in* $n$ *dimensional Euclidean space* $\mathbb{E}^n$. *For all* $P$ *in* $\mathbb{E}^n$,

$$\sum_{i=0}^{k}||P - v_i||_2^2 = \sum_{i=0}^{k}||G - v_i||_2^2 + (k+1)||P - G||_2^2, \tag{15}$$

*where* $G$ *is the centroid of* $v_i$, *will hold if it satisfies one of the following conditions:*

*(1)if* $k \geq n$ *and* $\mathbf{rank}(v_1 - v_0, v_2 - v_0, ..., v_k - v_0) = n$.

*(2)if* $k < n$ *and* $(v_1 - v_0, v_2 - v_0, ..., v_k - v_0)$ *are linearly independent.*

*(3)if* $v_i \sim N(\mathbf{0}, c^2 \cdot \mathbf{I}_n)$, *Eq.(15) holds with probability 1.*

*Proof.* (See Appendix K). $\qquad\square$

Let $P \in \{v_1, v_2, ..., v_N\}$. Since $v_i \sim N(\mathbf{0}, c^2 \cdot \mathbf{I})$, we can obtain that $a_i = P - v_i \sim N(\mathbf{0}, 2c^2 \cdot \mathbf{I})$ if $P \neq v_i$. According to the analysis in Section 3 and Theorem 4, we have

$$\sum_{i=1}^{n}||a_i||_2 \cong \sum_{i=1}^{n}||a_i||_2^2, \tag{16}$$

Next, we can prove $(k+1)||P - F||_2^2$ (**Fermat**) and $\sum_{i=1}^{N} ||P - v_i||_2$ (**GM**) are *approximately monotonic*, where $P \in \{v_1, v_2, ..., v_N\}$.

$$(k+1)||P - F||_2^2 \cong (k+1)||P - G||_2^2 \qquad \text{Since Eq. (13)}$$

$$= \sum_{i=1}^{N} ||P - v_i||_2^2 - \sum_{i=1}^{N} ||G - v_i||_2^2 \qquad \text{Since Theorem 5}$$

$$\cong \sum_{i=1}^{N} ||P - v_i||_2 - \sum_{i=1}^{N} ||G - v_i||_2^2 \qquad \text{Since Eq. (16)}$$

$$\cong \sum_{i=1}^{N} ||P - v_i||_2 \qquad (17)$$

The reason for the last equation is that $\sum_{i=1}^{N} ||G - v_i||_2^2$ is a constant for given $v_i$.

## H  PROOF OF THEOREM 2

**Proposition 4** (Stirling's formula). [6] *For big enough $x$ and $x \in \mathbb{R}^+$, we have an approximation of Gamma function:*

$$\Gamma(x+1) \approx \sqrt{2\pi x} \left(\frac{x}{e}\right)^x. \qquad (18)$$

**Proposition 5** (FKG inequality). *If $f$ and $g$ are increasing functions on $\mathbb{R}^n$  (Graham, 1983), we have*

$$\mathbb{E}(f)\mathbb{E}(g) \leq \mathbb{E}(fg). \qquad (19)$$

*Say that a function on $\mathbb{R}^n$ is increasing if it is an increasing function in each of its arguments.(i.e., for fixed values of the other arguments).*

**Proposition 6.** *Let $f(X, Y)$ is a two dimensional differentiable function. According to Taylor theorem (Hormander, 1983), we have*

$$f(X,Y) = f(\mathbb{E}(X), \mathbb{E}(Y)) + \sum_{cyc}(X - \mathbb{E}(X))\frac{\partial}{\partial X}f(\mathbb{E}(X), \mathbb{E}(Y)) + Remainder1, \qquad (20)$$

$$f(X,Y) = f(\mathbb{E}(X), \mathbb{E}(Y)) + \sum_{cyc}(X - \mathbb{E}(X))\frac{\partial}{\partial X}f(\mathbb{E}(X), \mathbb{E}(Y)) +$$
$$\frac{1}{2}\sum_{cyc}(X - \mathbb{E}(X))^T \frac{\partial^2}{\partial X^2}f(\mathbb{E}(X), \mathbb{E}(Y))(X - \mathbb{E}(X)) + Remainder2 \qquad (21)$$

**Lemma 1.** *Let $X$ and $Y$ are random variables. Then we have such an estimation*

$$\mathbf{Var}\left(\frac{X}{Y}\right) \approx \left(\frac{\mathbb{E}(X)}{\mathbb{E}(Y)}\right)^2 \left(\frac{\mathbf{Var}X}{\mathbb{E}(X)^2} + \frac{\mathbf{Var}Y}{\mathbb{E}(Y)^2} - 2\frac{\mathbf{Cov}(X, Y)}{\mathbb{E}(X)\mathbb{E}(Y)}\right). \qquad (22)$$

---

[6]en.wikipedia.org/wiki/Stirling'sapproximation

*Proof.* Let $f(X, Y) = X/Y$, according to the definition of variance, we have

$$\mathbf{Var}f(X, Y) = \mathbb{E}[f(X, Y) - \mathbb{E}(f(X, Y))]^2$$

$$\approx \mathbb{E}[f(X, Y) - \mathbb{E}\left\{f(\mathbb{E}(X), \mathbb{E}(Y)) + \sum_{cyc}(X - \mathbb{E}(X))\frac{\partial}{\partial X}f(\mathbb{E}(X), \mathbb{E}(Y))\right\}]^2$$

from Eq. (20)

$$= \mathbb{E}[f(X, Y) - f(\mathbb{E}(X), \mathbb{E}(Y)) - \sum_{cyc}\mathbb{E}(X - \mathbb{E}(X))\frac{\partial}{\partial X}f(\mathbb{E}(X), \mathbb{E}(Y))]^2$$

$$= \mathbb{E}[f(X, Y) - f(\mathbb{E}(X), \mathbb{E}(Y))]^2$$

$$\approx \mathbb{E}[\sum_{cyc}(X - \mathbb{E}(X))\frac{\partial}{\partial X}f(\mathbb{E}(X), \mathbb{E}(Y))]^2$$

from Eq. (20)

$$= 2\mathbf{Cov}(X, Y)\frac{\partial}{\partial X}f(\mathbb{E}(X), \mathbb{E}(Y))\frac{\partial}{\partial Y}f(\mathbb{E}(X), \mathbb{E}(Y)) + \sum_{cyc}[\frac{\partial}{\partial X}f(\mathbb{E}(X), \mathbb{E}(Y))]^2 \cdot \mathbf{Var}X$$

$$= 2\mathbf{Cov}(X, Y) \cdot \frac{1}{\mathbb{E}(Y)} \cdot \left(-\frac{\mathbb{E}(X)}{(\mathbb{E}(Y))^2}\right) + \frac{1}{(\mathbb{E}(Y))^2} \cdot \mathbf{Var}X + \frac{(\mathbb{E}X)^2}{(\mathbb{E}Y)^4} \cdot \mathbf{Var}Y$$

$$= \left(\frac{\mathbb{E}(X)}{\mathbb{E}(Y)}\right)^2 \left(\frac{\mathbf{Var}X}{\mathbb{E}(X)^2} + \frac{\mathbf{Var}Y}{\mathbb{E}(Y)^2} - 2\frac{\mathbf{Cov}(X, Y)}{\mathbb{E}(X)\mathbb{E}(Y)}\right).$$

$\square$

**Lemma 2.** *For big enough $x$ and $x \in \mathbb{R}^+$, we have*

$$\lim_{x \to +\infty} \left[\frac{\Gamma(\frac{x+1}{2})}{\Gamma(\frac{x}{2})}\right]^2 \cdot \frac{1}{x} = \frac{1}{2}. \tag{23}$$

*And*

$$\lim_{x \to +\infty} \frac{\Gamma(\frac{x}{2} + 1)}{\Gamma(\frac{x}{2})} - \left[\frac{\Gamma(\frac{x+1}{2})}{\Gamma(\frac{x}{2})}\right]^2 = \frac{1}{4}. \tag{24}$$

*Proof.*

$$\lim_{x \to +\infty} \left[\frac{\Gamma(\frac{x+1}{2})}{\Gamma(\frac{x}{2})}\right]^2 \cdot \frac{1}{x} \approx \lim_{x \to +\infty} \left(\frac{\sqrt{2\pi(\frac{x-1}{2})} \cdot (\frac{x-1}{2e})^{\frac{x-1}{2}}}{\sqrt{2\pi(\frac{x-2}{2})} \cdot (\frac{x-2}{2e})^{\frac{x-2}{2}}}\right)^2 \cdot \frac{1}{x} \qquad \text{from Proposition. 4}$$

$$= \lim_{x \to +\infty} \left(\frac{x-1}{x-2}\right) \cdot \frac{(\frac{x-1}{2e})^{x-2}}{(\frac{x-2}{2e})^{x-2}} \cdot \left(\frac{x-1}{2e}\right) \cdot \frac{1}{x}$$

$$= \lim_{x \to +\infty} \left(1 + \frac{1}{x-2}\right)^{x-2} \cdot \frac{x-1}{x-2} \cdot \frac{x-1}{2e} \cdot \frac{1}{x}$$

$$= \frac{1}{2}$$

on the other hand, we have

$$\lim_{x \to +\infty} \frac{\Gamma(\frac{x}{2} + 1)}{\Gamma(\frac{x}{2})} - \left[\frac{\Gamma(\frac{x+1}{2})}{\Gamma(\frac{x}{2})}\right]^2 = \lim_{x \to +\infty} \frac{x}{2} - \left(1 + \frac{1}{x-2}\right)^{x-2} \cdot \frac{x-1}{x-2} \cdot \frac{x-1}{2e}$$

$$= \lim_{x \to +\infty} \frac{x}{2e}\left(e - (1 + \frac{1}{x})^x\right)$$

$$= \frac{1}{2}\left(-\frac{\frac{1}{e}(-e)}{2}\right)$$

$$= \frac{1}{4}$$

$\square$

**Theorem 2** Let $X \sim N(\mathbf{0}, c^2 \cdot \mathbf{I}_n)$, we have

$$\mathbf{max} \left\{ \mathbf{Var}_X \left( \frac{\widehat{\ell}_2(X)}{\widehat{\ell}_1(X)} \right), \mathbf{Var}_X \left( \frac{\widehat{\ell}_1(X)}{\widehat{\ell}_2(X)} \right) \right\} \lesssim \frac{1}{n}.$$

where $\widehat{\ell}_1(X)$ denotes $\ell_1(X)/\mathbb{E}(\ell_1(X))$ and $\widehat{\ell}_2(X)$ denotes $\ell_2(X)/\mathbb{E}(\ell_2(X))$.

*Proof.* For the ratio $\widehat{\ell}_2(X)/\widehat{\ell}_1(X)$, we have

$$\mathbf{Var} \left( \frac{\widehat{\ell}_2(X)}{\widehat{\ell}_1(X)} \right) = \left( \frac{\mathbb{E}(\ell_1(X))}{\mathbb{E}(\ell_2(X))} \right)^2 \mathbf{Var} \left( \frac{\widehat{\ell}_2(X)}{\widehat{\ell}_1(X)} \right)$$

$$\approx \left( \frac{\mathbb{E}(\ell_1(X))}{\mathbb{E}(\ell_2(X))} \right)^2 \left( \frac{\mathbb{E}(\ell_2(X))}{\mathbb{E}(\ell_1(X))} \right)^2 \left( \frac{\mathbf{Var}\ell_2(X)}{\mathbb{E}(\ell_2(X))^2} + \frac{\mathbf{Var}\ell_1(X)}{\mathbb{E}(\ell_1(X))^2} - 2\frac{\mathbf{Cov}(\ell_2(X), \ell_1(X))}{\mathbb{E}(\ell_2(X))\mathbb{E}(\ell_1(X))} \right)$$

$$\text{from Lemma. 1}$$

$$\leq \left( \frac{\mathbf{Var}\ell_2(X)}{\mathbb{E}(\ell_2(X))^2} + \frac{\mathbf{Var}\ell_1(X)}{\mathbb{E}(\ell_1(X))^2} \right). \qquad \text{from Proposition. 5}$$

similarly, we also have

$$\mathbf{Var} \left( \frac{\widehat{\ell}_1(X)}{\widehat{\ell}_2(X)} \right) \leq \left( \frac{\mathbf{Var}\ell_2(X)}{\mathbb{E}(\ell_2(X))^2} + \frac{\mathbf{Var}\ell_1(X)}{\mathbb{E}(\ell_1(X))^2} \right). \qquad (25)$$

Therefore,

$$\mathbf{max} \left\{ \mathbf{Var}_X \left( \frac{\widehat{\ell}_2(X)}{\widehat{\ell}_1(X)} \right), \mathbf{Var}_X \left( \frac{\widehat{\ell}_1(X)}{\widehat{\ell}_2(X)} \right) \right\} \leq \left( \frac{\mathbf{Var}\ell_2(X)}{\mathbb{E}(\ell_2(X))^2} + \frac{\mathbf{Var}\ell_1(X)}{\mathbb{E}(\ell_1(X))^2} \right)$$

$$= \frac{2\sigma^2 \left[ \frac{\Gamma(\frac{n}{2}+1)}{\Gamma(\frac{n}{2})} - \frac{\Gamma(\frac{n+1}{2})^2}{\Gamma(\frac{n}{2})^2} \right]}{(\sqrt{2}\sigma \cdot \frac{\Gamma(\frac{n+1}{2})}{\Gamma(\frac{n}{2})})^2} + \frac{\sigma^2 \left( 1 - \frac{2}{\pi} \right) n}{(n \cdot \sigma\sqrt{2/\pi})^2}$$

$$\text{from Proposition. 3 and 2}$$

$$\approx \left( \frac{1}{2n} + (\frac{\pi}{2} - 1)\frac{1}{n} \right) \qquad \text{from Lemma 2}$$

$$= \frac{\pi - 1}{2n}$$

$\square$

Because the approximation is widely used in the proof of Theorem 1, it is necessary to verify it numerically. As shown in Fig. 17, we use ResNet56 on Cifar100 and ResNet110 on Cifar10 respectively to verify Theorem 1. From Fig. 17, we find that the estimationn of Theorem 1 is reliable, *i.e.*, the estimation $O(\frac{1}{n})$ for $\mathbf{max} \left\{ \mathbf{Var}_X \left( \frac{\widehat{\ell}_2(X)}{\widehat{\ell}_1(X)} \right), \mathbf{Var}_X \left( \frac{\widehat{\ell}_1(X)}{\widehat{\ell}_2(X)} \right) \right\}$ is appropriate.

## I   PROOF OF THEOREM 3

**Proposition 7.** *Let $L_p^{(\alpha)}(x)$ denotes generalized Laguerre function, and it have following properties:*

$$\frac{\partial^n}{\partial x^n} L_p^{(\alpha)} = (-1)^n L_{p-n}^{(\alpha+n)}(x), \qquad (26)$$

*and for $\alpha > 0$,*

$$L_{-\frac{1}{2}}^{(\alpha)}(x) > 0. \qquad (27)$$

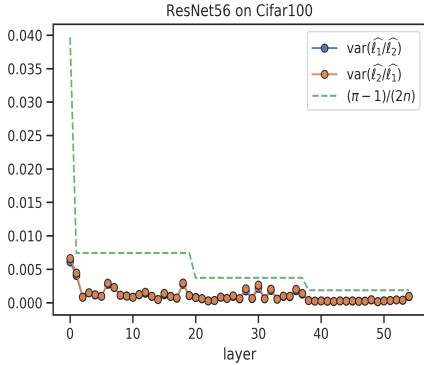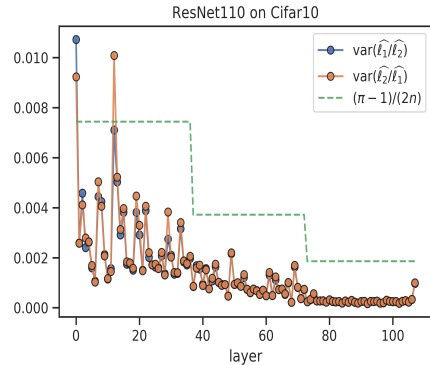

Figure 17: The approximation of **Theorem 2**: (Left) the example about ResNet56; (Right) the example about ResNet110.

**Theorem 3.** Let random variable $v_i \in \mathbb{R}^k$. They are i.i.d and follow normal distribution $N(\mathbf{0}, \sigma^2 \mathbf{I}_k)$. For $F$ in $\mathbb{R}^k$, we have

$$\mathbf{argmin}_F \left\{ \mathbb{E}_{v_i \sim N(\mathbf{0}, \sigma^2 \mathbf{I}_k)} \sum_{i=1}^{n} ||F - v_i||_2 \right\} = \mathbf{0}.$$

*Proof.* Let $w_i = F - v_i$ and we have $w_i \sim N(F, \sigma^2 \mathbf{I}_k)$, then

$$\mathbb{E}_{v_i \sim N(\mathbf{0}, \sigma^2 \mathbf{I}_k)} \sum_{i=1}^{n} ||F - v_i||_2 = \sum_{i=1}^{n} \mathbb{E}_{v_i \sim N(\mathbf{0}, \sigma^2 \mathbf{I}_k)} ||F - v_i||_2$$

$$= \sum_{i=1}^{n} \mathbb{E}_{w_i \sim N(F, \sigma^2 \mathbf{I}_k)} ||w_i||_2$$

$$= n \cdot \sigma^2 \sqrt{\frac{\pi}{2}} \cdot L_{\frac{1}{2}}^{(\frac{k}{2} - 1)} \left( -\frac{||F||_2^2}{2\sigma^2} \right)$$

The reason for the last equation is that $||w_i||_2$ follows scaled noncentral chi distribution[7] when $w_i \sim N(F, \sigma^2 \mathbf{I}_k)$. Let $T(x) = L_{\frac{1}{2}}^{(\frac{k}{2} - 1)} \left( -\frac{x^2}{2\sigma^2} \right)$, we calculate the minimum of $T(x)$. From Eq. (26),

$$\frac{d}{dx} T(x) = \frac{x}{\sigma^2} \cdot L_{-\frac{1}{2}}^{(\frac{k}{2})} \left( -\frac{x^2}{2\sigma^2} \right). \tag{28}$$

Since Eq. (27), we find that $\frac{d}{dx} T(x) > 0$ when $x > 0$ and if $x \leq 0$, then $\frac{d}{dx} T(x) \leq 0$. It means that $T(x)$ gets the minimizer at $||F||_2 = 0$, *i.e.*, $F = \mathbf{0}$.

$\square$

## J  PROOF OF THEOREM 4

**Lemma 3.** *For two random variables $X, Y \in \mathbb{R}^k$ follow $N(\mathbf{0}, c^2 \cdot \mathbf{I}_k)$ and they are i.i.d. When $k$ is large enough, we have:*

$$\mathbb{E} \left( \frac{(||X||_2^2 - ||Y||_2^2)^2}{2||X||_2 \cdot ||Y||_2} \right) \approx 2c^2 + \frac{4c^2 k + 1}{2k^2}, \tag{29}$$

*and*

$$\mathbf{Var} \left( \frac{(||X||_2^2 - ||Y||_2^2)^2}{2||X||_2 \cdot ||Y||_2} \right) \lesssim 8c^4 + \frac{16c^4 k + c^2}{k^2}, \tag{30}$$

---

[7]Survey of simple,continuous,uniariate probability distribution and Wikipredia.

*Proof.* According to **Proposition 3** and **Lemma 2**, it is easy to know (similar method in Eq.(86)), when $k$ is large enough, that

$$\mathbb{E}\left(2||X||_2 \cdot ||Y||_2\right) = 2c^2k, \quad \mathbf{Var}\left(2||X||_2 \cdot ||Y||_2\right) = c^2 + 4c^4k, \tag{31}$$

and

$$\mathbb{E}\left((||X||_2^2 - ||Y||_2^2)^2\right) = 4c^4k, \quad \mathbf{Var}\left((||X||_2^2 - ||Y||_2^2)^2\right) = 16c^8(2k^2 + 3k). \tag{32}$$

Since Lemma 1, we have an estimation

$$\mathbf{Var}\left(\frac{(||X||_2^2 - ||Y||_2^2)^2}{2||X||_2 \cdot ||Y||_2}\right) \leq \left(\frac{\mathbb{E}(||X||_2^2 - ||Y||_2^2)^2}{\mathbb{E}2||X||_2 \cdot ||Y||_2}\right)^2 \left(\frac{\mathbf{Var}(||X||_2^2 - ||Y||_2^2)^2}{\mathbb{E}(||X||_2^2 - ||Y||_2^2)^2} + \frac{\mathbf{Var}(2||X||_2 \cdot ||Y||_2)^2)}{\mathbb{E}(2||X||_2 \cdot ||Y||_2)^2}\right)$$

$$\approx \left(\frac{4c^4k}{2c^2k}\right)^2 \cdot \left(\frac{c^2 + 4c^4k}{4c^4k} + \frac{16c^8(2k^2 + 3k)}{16c^8k^2}\right) \qquad \text{Since Eq.(31) and Eq.(32)}$$

$$= 8c^4 + \frac{16c^4k + c^2}{k^2}.$$

From Eq.(21) and **Lemma 1**, we also can obtain an estimation of $\mathbb{E}(\mathbf{A}/\mathbf{B})$, where $\mathbf{A}$ and $\mathbf{B}$ are two random variables. *i.e.*,

$$\mathbb{E}\left(\frac{\mathbf{A}}{\mathbf{B}}\right) \approx \frac{\mathbb{E}\mathbf{A}}{\mathbb{E}\mathbf{B}} + \mathbf{Var}(\mathbf{B}) \cdot \frac{\mathbb{E}\mathbf{A}}{(\mathbb{E}\mathbf{B})^3}. \tag{33}$$

Therefore,

$$\mathbb{E}\left(\frac{(||X||_2^2 - ||Y||_2^2)^2}{2||X||_2 \cdot ||Y||_2}\right) \approx \frac{\mathbb{E}(||X||_2^2 - ||Y||_2^2)^2}{\mathbb{E}2||X||_2 \cdot ||Y||_2} + \mathbf{Var}(2||X||_2 \cdot ||Y||_2) \cdot \frac{\mathbb{E}(||X||_2^2 - ||Y||_2^2)^2}{(\mathbb{E}2||X||_2 \cdot ||Y||_2)^3}$$

$$\text{Since Eq.(33)}$$

$$\approx \frac{4c^4k}{2c^2k} + \frac{4c^4k}{8c^6k^3} \cdot (c^2 + 4c^4k) \qquad \text{Since Eq.(31) and Eq.(32)}$$

$$= 2c^2 + \frac{4c^2k + 1}{2k^2}.$$

$$\square$$

Note that, the approximation is widely used in the proof of Eq.(29) and Eq.(30). Hence, it is also necessary to verify it numerically. As shown in Fig. 18, the estimation is appropriate. According to **Lemma** 3, the mathematical expectation and variance of the ratio of $(||X||_2^2 - ||Y||_2^2)^2$ and $2||X||_2 \cdot ||Y||_2$ are both close to 0 when $k$ is large enough and $c$ is small enough. that is,

$$2(||X||_2 \cdot ||Y||_2) \gg (||X||_2^2 - ||Y||_2^2)^2. \tag{34}$$

By the way, the convolutional filters easily meet the condition that $k$ is large enough.

**Theorem 4.** For $n$ random variables $a_i \in \mathbb{R}^k$ follow $N(\mathbf{0}, c^2 \cdot \mathbf{I}_k)$. When $k$ is large enough, we have such an estimation:

$$\mathbf{Var}_{a_i} \frac{F_1(a_i)}{F_2(a_i)} \approx \frac{1}{2nk}, \quad \mathbf{Var}_{a_i} \frac{F_2(a_i)}{F_1(a_i)} \approx \frac{1}{2nk}.$$

where $F_1(a_i) = \sum_{i=1}^{n} ||a_i||_2 / \mathbb{E}(\sum_{i=1}^{n} ||a_i||_2)$ and $F_2(a_i) = \sum_{i=1}^{n} ||a_i||_2^2 / \mathbb{E}(\sum_{i=1}^{n} ||a_i||_2^2)$.

*Proof.* Since Eq. (9) and Eq. (10), we have

$$\mathbf{Var}_{a_i} \frac{F_1(a_i)}{F_2(a_i)} = \left(\frac{nc^2k}{nc\sqrt{k}}\right)^2 \cdot \mathbf{Var}_{a_i}\left(\frac{\sum_{i=1}^{n} ||a_i||_2}{\sum_{i=1}^{n} ||a_i||_2^2}\right). \tag{35}$$

and

$$\mathbf{Var}_{a_i} \frac{F_2(a_i)}{F_1(a_i)} = \left(\frac{nc\sqrt{k}}{nc^2k}\right)^2 \cdot \mathbf{Var}_{a_i}\left(\frac{\sum_{i=1}^{n} ||a_i||_2^2}{\sum_{i=1}^{n} ||a_i||_2}\right). \tag{36}$$

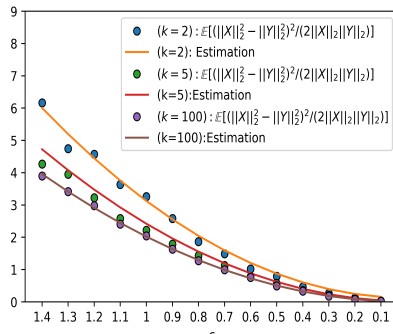 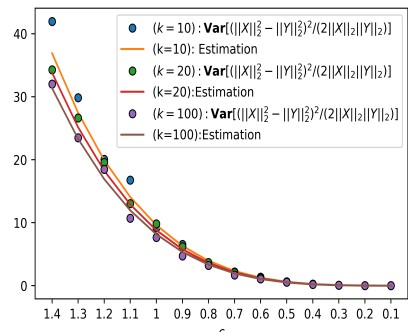

Figure 18: (Left) The numerical verification of Eq.(29) and (Right) The numerical verification of Eq.(30). $X$ and $Y$ follow $N(\mathbf{0}, c^2 \cdot I_k)$.

According to Lagrange's identity, we have

$$
\left(\sum_{i=1}^n ||a_i||_2^2\right)\left(\sum_{i=1}^n 1\right) = \left(\sum_{i=1}^n ||a_i||_2\right)^2 + \sum_{1 \leq i < j \leq n}(||a_i||_2^2 - ||a_j||_2^2)^2
$$

$$
= \sum_{i=1}^n ||a_i||_2^2 + \sum_{1 \leq i < j \leq n}(||a_i||_2 \cdot ||a_j||_2) + 2\sum_{1 \leq i < j \leq n}(||a_i||_2^2 - ||a_j||_2^2)^2
$$

$$
\approx \sum_{i=1}^n ||a_i||_2^2 + 2\sum_{1 \leq i < j \leq n}(||a_i||_2 \cdot ||a_j||_2) \qquad \text{Since Eq. (34)}
$$

$$
= \left(\sum_{i=1}^n ||a_i||_2\right)^2
$$

so we have

$$
\mathbf{Var}_{a_i \sim N(\mathbf{0},c^2\cdot \mathbf{I}_k)}\frac{\sum_{i=1}^n ||a_i||_2}{\sum_{i=1}^n ||a_i||_2^2} \approx \mathbf{Var}_{a_i \sim N(\mathbf{0},c^2\cdot \mathbf{I}_k)}\frac{n}{\sum_{i=1}^n ||a_i||_2} \qquad (37)
$$

By central limit theorem, we have $\sqrt{n}(\frac{1}{n}\sum_{i=1}^n ||a_i||_2 - \mu) \sim N(0, \sigma^2)$. And let $g(x) = \frac{1}{x}$, we can use Delta method[8] to find the distribution of $g(\frac{1}{n}\sum_{i=1}^n ||a_i||_2)$:

$$
\sqrt{n}\left(g(\frac{\sum_{i=1}^n ||a_i||_2}{n}) - g(\mu))\right) \sim N(0, \sigma^2 \cdot [g\prime(\mu)]^2) = N(0, \sigma^2 \cdot \frac{1}{\mu^4}). \qquad (38)
$$

where $\mu$ and $\sigma^2$ denote the mean and variance of $||a_i||_2$ respectively. From Eq. (37), we have

$$
\mathbf{Var}_{a_i \sim N(\mathbf{0},c^2\cdot \mathbf{I}_k)}\frac{\sum_{i=1}^n ||a_i||_2}{\sum_{i=1}^n ||a_i||_2^2} \approx \mathbf{Var}_{a_i \sim N(\mathbf{0},c^2\cdot \mathbf{I}_k)}\frac{n}{\sum_{i=1}^n ||a_i||_2}
$$

$$
= \sigma^2 \cdot \frac{1}{\mu^4 \cdot n} \qquad \text{Since Eq. (38)}
$$

$$
= 2c^2\left[\frac{\Gamma(\frac{k}{2}+1)}{\Gamma(\frac{k}{2})} - \frac{\Gamma(\frac{k+1}{2})^2}{\Gamma(\frac{k}{2})^2}\right] \cdot \frac{1}{(\sqrt{2}c \cdot \frac{\Gamma(\frac{k+1}{2})}{\Gamma(\frac{k}{2})})^4 \cdot n}
$$

$$
\qquad\qquad\qquad\qquad\qquad\qquad\qquad\qquad \text{Since Eq. (9) and Eq. (10)}
$$

$$
= \frac{1}{2c^2 \cdot nk^2} \qquad \text{Since Lemma. 2}
$$

---

[8] https://en.wikipedia.org/wiki/Delta_method

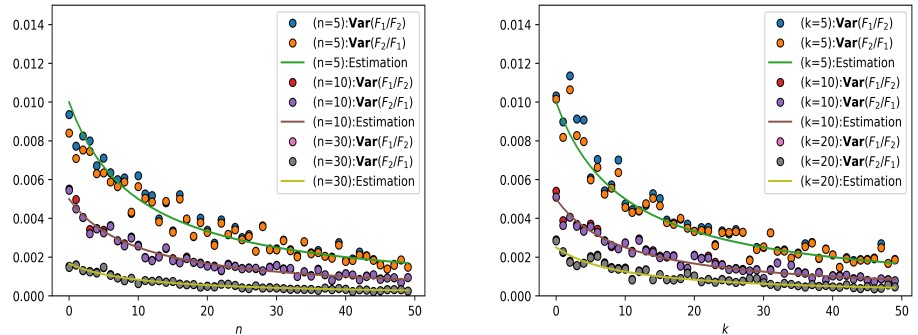

Figure 19: A numerical verification of **Theorem 4**, where $F_1 = \sum_{i=1}^n ||a_i||_2 / \mathbb{E}(\sum_{i=1}^n ||a_i||_2)$ and $F_2 = \sum_{i=1}^n ||a_i||_2^2 / \mathbb{E}(\sum_{i=1}^n ||a_i||_2^2)$. $a_i$ follow $N(\mathbf{0}, 0.01^2 \cdot I_k)$.

Since Eq. (35), we have

$$\mathbf{Var}_{a_i} \frac{F_1(a_i)}{F_2(a_i)} = \left(\frac{nc^2 k}{nc\sqrt{k}}\right)^2 \cdot \mathbf{Var}_{a_i} \left(\frac{\sum_{i=1}^n ||a_i||_2}{\sum_{i=1}^n ||a_i||_2^2}\right) \approx \frac{1}{2nk}. \tag{39}$$

Similar to Eq. (37),

$$\mathbf{Var}_{a_i \sim N(\mathbf{0}, c^2 \cdot \mathbf{I}_k)} \frac{\sum_{i=1}^n ||a_i||_2^2}{\sum_{i=1}^n ||a_i||_2} \approx \mathbf{Var}_{a_i \sim N(\mathbf{0}, c^2 \cdot \mathbf{I}_k)} \frac{\sum_{i=1}^n ||a_i||_2}{n} \tag{40}$$

$$
\begin{aligned}
\mathbf{Var}_{a_i \sim N(\mathbf{0}, c^2 \cdot \mathbf{I}_k)} \frac{\sum_{i=1}^n ||a_i||_2^2}{\sum_{i=1}^n ||a_i||_2} &\approx \mathbf{Var}_{a_i \sim N(\mathbf{0}, c^2 \cdot \mathbf{I}_k)} \frac{\sum_{i=1}^n ||a_i||_2}{n} && \text{Similar to Eq. (37)} \\
&= \sigma^2 \cdot \frac{1}{n} && \text{Since central limit theorem} \\
&= 2c^2 \left[\frac{\Gamma(\frac{k}{2}+1)}{\Gamma(\frac{k}{2})} - \frac{\Gamma(\frac{k+1}{2})^2}{\Gamma(\frac{k}{2})^2}\right] \cdot \frac{1}{n} && \text{Since Eq. (10)} \\
&= \frac{c^2}{2n} && \text{Since Lemma. 2}
\end{aligned}
$$

Since Eq. (36), we have

$$\mathbf{Var}_{a_i} \frac{F_2(a_i)}{F_1(a_i)} = \left(\frac{nc\sqrt{k}}{nc^2 k}\right)^2 \cdot \mathbf{Var}_{a_i} \left(\frac{\sum_{i=1}^n ||a_i||_2^2}{\sum_{i=1}^n ||a_i||_2}\right) \approx \frac{1}{2nk}. \tag{41}$$

From Eq.(39) and Eq.(41), **Theorem 4** holds.

$$\square$$

In Fig. 19, we also show a numerical verification of **Theorem 4**.

## K  PROOF OF THEOREM 5

**Proposition 8.** *For a $n \times m$ random matrix $(a_{ij})_{n \times m}$, where $a_{ij} \sim N(0, \sigma^2)$. And Eq. (8) holds with probability 1.*

$$\mathbf{rank}((a_{ij})_{n \times m}) = \mathbf{min}(m, n). \tag{42}$$

**Lemma 4.** *Let $v_0, v_1, ..., v_k$ be the $k + 1$ vectors in $n$ dimensional Euclidean space $V$ and $k \leq n$. If $\mathbf{rank}(v_1 - v_0, v_2 - v_0, ..., v_k - v_0) = n$, then $\forall x \in V, \exists \lambda_i (0 \leq i \leq k)$, s.t.*

$$x = \sum_{i=0}^{k} \lambda_i \cdot v_i, \tag{43}$$

*and $\sum_{i=0}^{k} \lambda_i = 1$. We call $\lambda = (\lambda_0, \lambda_1, ..., \lambda_k)$ the generalized barycentric coordinate with respect to $(v_0, v_1, ..., v_k)$. (In general, barycentric coordinate is a concept in Polytope)*

*Proof.* Note that $v_i$ is the element of $n$ dimensional linear space $V$ and $\mathbf{rank}(v_1 - v_0, v_2 - v_0, ..., v_k - v_0) = n$. It means $(v_1 - v_0, v_2 - v_0, ..., v_k - v_0)$ form a set of basis in the linear space $V$. $\forall x \in V$, $x - v_0$ can be expressed linearly by them, *i.e.*,$\exists t_i (1 \leq i \leq k)$ s.t.

$$x = v_0 + \sum_{i=1}^{k} t_i (v_i - v_0)$$

$$= (1 - \sum_{i=1}^{k} t_i) v_0 + \sum_{i=1}^{k} t_i v_i.$$

Let $\lambda_0 = (1 - \sum_{i=1}^{k} t_i)$ and $\lambda_i = t_i (1 \leq i \leq k)$, Lemma 4 holds. $\square$

**Lemma 5.** *Let $v_0, v_1, ..., v_k$ be the $k + 1$ vectors in $n$ dimensional Euclidean space $V$. $\forall a, b \in V$, and the generalized barycentric coordinate of $a, b$ with respect to $(v_0, v_1, ..., v_k)$ are $\lambda = (\lambda_0, \lambda_1, ..., \lambda_k)^T$ and $\mu = (\mu_0, \mu_1, ..., \mu_k)^T$, respectively. Then*

$$||a - b||_2^2 = (\lambda - \mu)^T D(\lambda - \mu), \tag{44}$$

*where $D = (-\frac{1}{2} d_{ij})_{(k+1) \times (k+1)}$, and $d_{ij} = ||v_i - v_j||_2^2$.*

*Proof.* Since Lemma 4, let $R = [v_0, v_1, ..., v_k]_{n \times (k+1)}$, and we have $a = R\lambda$ and $b = R\mu$. Moreover,

$$||a - b||_2^2 = (a - b)^T (a - b) \tag{45}$$

$$= [R(\lambda - \mu)]^T [R(\lambda - \mu)] \tag{46}$$

$$= (\lambda - \mu)^T R^T R(\lambda - \mu). \tag{47}$$

Note that, for $D = (-\frac{1}{2} d_{ij})_{(k+1) \times (k+1)}$,

$$-\frac{1}{2} d_{ij} = -\frac{1}{2} (v_i - v_j)^T (v_i - v_j) \tag{48}$$

$$= v_i^T v_j - \frac{1}{2} (v_i^T v_i + v_j^T v_j). \tag{49}$$

So we have $D = R^T R - \frac{1}{2} \left( (v_i^T v_i + v_j^T v_j)_{(k+1) \times (k+1)} \right)$. It can be further simplified to $D = R^T R - \frac{1}{2} (V\alpha^T + \alpha V^T)$, where $V = (v_0^T v_0, ..., v_k^T v_k)^T$ and $\alpha = (1, ..., 1)^T$. So

$$||a - b||_2^2 = (\lambda - \mu)^T R^T R(\lambda - \mu) \tag{50}$$

$$= (\lambda - \mu)^T (D + \frac{1}{2} (V\alpha^T + \alpha V^T))(\lambda - \mu) \tag{51}$$

$$= (\lambda - \mu)^T D(\lambda - \mu) + \frac{1}{2} (\lambda - \mu)^T (V\alpha^T + \alpha V^T)(\lambda - \mu), \tag{52}$$

therefore, we only need to prove $(\lambda - \mu)^T (V\alpha^T + \alpha V^T)(\lambda - \mu) = 0$. From Lemma 4, we have $\alpha^T (\lambda - \mu) = (\lambda - \mu)^T \alpha = 0$ and the Lemma 5 holds.

$\square$

**Definition 1** (Ultra dimension). *For a set $U$ composed of vectors in a $n$ dimensional linear space $V$, we define $\widehat{\dim}(U)$ as the Ultra dimension of $U$. The definition is that if $U$ has $k$ linearly independent vectors and there are no more, then $\widehat{\dim}(U) = k$.*

In fact, if $U$ is a linear subspace in $V$, then the Ultra dimension and the dimensions of the linear subspace are equivalent. If $U$ is a linear manifold, $U = \{x + v_0 | x \in W\}$, where $v_0$ and $W$ are non-zero vectors and linear subspaces in $V$, respectively. And $\dim(W) = r$. Then

$$\widehat{\dim}(U) = \begin{cases} r, & v_0 \in W \\ r + 1, & v_0 \notin W \end{cases} \tag{53}$$

In other words, $\widehat{\dim}(U) \geq \widehat{\dim}(W)$ always holds.

**Lemma 6.** *For arbitrary $k$ ($1 \leq k \leq n-1$), let $a_1, a_2, ..., a_k$ be $k$ linearly independent vectors in $n$ dimensional linear space $V$. Consider one $n-1$ dimensional linear subspace $W$ in $V$ and a non-zero vector $v_0$ in $V$. They form a linear manifold $P = \{v_0 + \alpha | \alpha \in W\}$. If $a_1, a_2, ..., a_k$ do not all belong to $P$, then there must exist $n - k$ vectors $p_1, p_2, ..., p_{n-k}$ from $P$, s.t $(a_1, a_2, ..., a_k, p_1, p_2, ..., p_{n-k})$ are a set of basis for the linear space $V$.*

*Proof.* we use mathematical induction. First, show that the Lemma 6 holds for $n - k = 1$. it means we need to find a vector $p_1 \in P$ s.t. $a_1, a_2, ..., a_k, p_1$ linearly independent. If $p_1$ does not exist, then $\forall p \in P$ would be linearly represented by $a_1, a_2, ..., a_k$. In other word,

$$P \subset L = \mathbf{span}(a_1, a_2, ..., a_k), \tag{54}$$

① For the linear manifold $P$, if $v_0 \in W$. This means that $P$ is equal to the linear subspace $W$. Since Eq. (54), we have $W \subset L$ and $\widehat{\dim}(W) = \widehat{\dim}(L)$. Hence, $P = W = L$. However, $a_1, a_2, ..., a_k$ do not all belong to $P$, a contradiction.

② For the linear manifold $P$, if $v_0 \notin W$, then $\widehat{\dim}(P) = n$. Because $v_0 \notin W$, that is, $v_0$ cannot be represented by a set of basis of $W$. In other words, $v_0$ and a set of basis of $W$ are linearly independent. However, the dimension of $W$ is $n - 1$, hence $\widehat{\dim}(P) = n$. From Eq. (54), we have $P \subset L$, so

$$n = \widehat{\dim}(P) \leq \widehat{\dim}(L) = k = n - 1, \tag{55}$$

a contradiction. Therefore, Lemma 6 holds for $n - k = 1$. Assume the induction hypothesis that Lemma 6 is true when $n - k = l$, where $1 \leq l$. when $n - k = l + 1$, *i.e.*, $k = n - (l + 1)$, we also can find a vector $p_1 \in P$ s.t. $a_1, a_2, ..., a_k, p_1$ linearly independent. Otherwise, $\forall p \in P$ would be linearly represented by $a_1, a_2, ..., a_k$. Similarly, we have Eq. (54). Note that, from Definition 1, $\widehat{\dim}(P) \geq n - 1$, hence

$$n - 1 \leq \widehat{\dim}(P) \leq \widehat{\dim}(L) = k = n - (l + 1). \tag{56}$$

a contradiction. At this time, we have $k + 1 = n - (l + 1) + 1 = n - l$ vectors $a_1, a_2, ..., a_k, p_1$ which are not all on $P$. Note that $n - (n - l) = l$, using the induction hypothesis, the Lemma 6 also holds for $n - k = l$. In summary, Lemma 6 holds.

$\square$

**Theorem 5.** Let $v_0, v_1, ..., v_k$ be the $k + 1$ vectors in $n$ dimensional Euclidean space $\mathbb{E}^n$. For all $P$ in $\mathbb{E}^n$,

$$\sum_{i=0}^{k} ||P - v_i||_2^2 = \sum_{i=0}^{k} ||G - v_i||_2^2 + (k + 1)||P - G||_2^2.$$

where $G$ is the centroid of $v_i$, will hold if it satisfies one of the following conditions:

(1)if $k \geq n$ and $\mathbf{rank}(v_1 - v_0, v_2 - v_0, ..., v_k - v_0) = n$.

(2)if $k < n$ and $(v_1 - v_0, v_2 - v_0, ..., v_k - v_0)$ are linearly independent.

(3)if $v_i \sim N(\mathbf{0}, c \cdot \mathbf{I}_n)$, Eq.(15) holds with probability 1 where $c$ is a constant.

*Proof.* **For Theorem 5 (1)**. From Lemma 4, $\forall P \in E^n$, $\exists \gamma = (\gamma_0, ..., \gamma_k)$, s.t. $P$ can be represented by $\sum_{i=0}^{k} \gamma_i v_i$, where $\sum_{i=0}^{k} \gamma_i = 1$. In fact, for each $v_i$, it also can be respresented by $\sum_{j=0}^{k} \beta_{ij} v_i$, where $\sum_{i=0}^{k} \beta_{ij} = 1$. We just take $(\beta_{i0}, \beta_{i1}, ..., \beta_{ik})$ as one of the standard orthogonal basis $\epsilon_i = (0, 0, ..., 1_i, ...0)$. According to lemma 5,

$$||P - v_i||_2^2 = (\gamma - \epsilon_i)^T D (\gamma - \epsilon_i) \tag{57}$$

$$= \gamma^T D \gamma - 2\gamma^T D \epsilon_i + \epsilon_i^T D \epsilon_i \tag{58}$$

$$= \gamma^T D \gamma - 2\gamma^T D \epsilon_i. \tag{59}$$

The final equation is because the diagonal elements of the matrix are all 0. On the other hand, we have

$$||G - v_i||_2^2 = (\frac{1}{k+1} \sum_{i=0}^{k} \epsilon_i - \epsilon_i)^T D (\frac{1}{k+1} \sum_{i=0}^{k} \epsilon_i - \epsilon_i) \tag{60}$$

$$= \frac{1}{(k+1)^2} \alpha^T D \alpha - \frac{2}{k+1} \alpha^T D \epsilon_i + \epsilon_i^T D \epsilon_i \tag{61}$$

$$= \frac{1}{(k+1)^2} \alpha^T D \alpha - \frac{2}{k+1} \alpha^T D \epsilon_i, \tag{62}$$

where $\alpha = \sum_{i=0}^{k} \epsilon_i$, *i.e.*,$\alpha = (1, 1, ..., 1)$. Next, we consider $||P - G||_2^2$.

$$||P - G||_2^2 = (\gamma - \frac{1}{k+1} \alpha)^T D (\gamma - \frac{1}{k+1} \alpha) \tag{63}$$

$$= \gamma^T D \gamma + \frac{1}{(k+1)^2} \alpha^T D \alpha - \frac{2}{k+1} \gamma^T D \alpha. \tag{64}$$

In summary, we have

$$\sum_{i=0}^{k} ||P - v_i||_2^2 - ||G - v_i||_2^2 = (k+1)\gamma^T D \gamma - 2\gamma^T D \alpha + \frac{1}{k+1} \alpha^T D \alpha \tag{65}$$

$$= (k+1)||P - G||_2^2 \tag{66}$$

Therefore, Theorem 5 (1) holds.

**For Theorem 5 (2)**. Next, we prove the case of $k < n$. Obviously, Lemma 4 does not hold. We consider about such a linear space $W_1 = \mathbf{span}(P - G)$, *i.e.*, a linear space expanded by $P - G$, and its orthogonal complement $W_1^{\perp}$ (in $E^n$). Since dimension formula from linear space, it is easy to konw that $\mathbf{dim}(W_1^{\perp}) = n - 1$.

Two linear manifolds $T_1$ and $T_2$ are constructed as follows,

$$T_1 = \{x + G | x \in W_1^{\perp}\} \tag{67}$$

$$T_2 = \{x + G - v_0 | x \in W_1^{\perp}\} \tag{68}$$

$\forall v_i \in T_1$, we have $(v_i - G)^T (P - G) = 0$, Furthermore,

$$||P - v_i||_2^2 = ||v_i - G||_2^2 + ||P - G||_2^2. \tag{69}$$

It is easy to know that $G - v_0$ is not 0. If $v_1 - v_0, ..., v_k - v_0$ are all belong to $T_2$, it means $v_1, .., v_k$ are all in $T_1$. Hence, we have Eq. (69). By summing both sides of Eq. (69) for $i$, it is obvious find that Theorem 5 (2) holds. If $v_1 - v_0, ..., v_k - v_0$ are not all belong to $T_2$, since Lemma 6, there are $n - k$ vectors $p_1 - v_0, p_2 - v_0, .., p_{n-k} - v_0$ from $T_2$ s.t. they and $v_1 - v_0, ..., v_k - v_0$ are linearly independent, where $p_i$ obviously belongs to manifold $T_1$.

At the same time, we have $2G - p_i \in T_1$, we can also construct $n - k$ new vectors $2G - p_i - v_0 \in T_2$ and calculate the rank that

$$\mathbf{rank}(v_1 - v_0, ..., v_k - v_0, p_1 - v_0, ..., p_{n-k} - v_0, 2G - p_1 - v_0, ..., 2G - p_{n-k} - v_0)$$

$$= \mathbf{rank}(v_1 - v_0, ..., v_k - v_0, p_1 - v_0, ..., p_{n-k} - v_0, 2(G - v_0), ..., 2(G - v_0)) \tag{70}$$

$$= \mathbf{rank}(v_1 - v_0, ..., v_k - v_0, p_1 - v_0, ..., p_{n-k} - v_0, 0, ..., 0) \tag{71}$$

$$= n \tag{72}$$

The reason of the final equation is that $\sum_{i=1}^{k}(v_i - v_0) = (k+1)(G - v_0)$. Note that there are a total of $k + (n-k) + (n-k) = n + (n-k) \geq n$ vectors, meets the lemma 4 condition. For the convenience of description, we define

$$L_i^{(1)} = v_i, (0 \leq i \leq k), \tag{73}$$

$$L_i^{(2)} = p_i, (1 \leq i \leq n-k), \tag{74}$$

$$L_i^{(3)} = 2G - p_i, (1 \leq i \leq n-k). \tag{75}$$

And their centroid is

$$G' = \frac{1}{2n-k+1}\left(\sum_{i=0}^{k} v_i + \sum_{i=1}^{n-k}(L_i^{(2)} + L_i^{(3)})\right) \tag{76}$$

$$= \frac{1}{2n-k+1}((k+1)G + 2(n-k)G) \tag{77}$$

$$= G \tag{78}$$

That is, the newly added vector does not change the centroid of $v_i$. On the other hand, since both $L_i^{(2)}$ and $L_i^{(3)}$ are in the linear manifold $T_1$, and it meets the conditions of the Eq.(69). Similar to the derivation in the Theorem 5 (1), we have

$$(2n-k+1)||P - G||_2^2 = \sum_{t=L_i^{(1)}, L_i^{(2)}, L_i^{(3)}} \left(||P - t||_2^2 - ||G - t||_2^2\right) \tag{79}$$

$$= \sum_{i=0}^{k}\left(||P - v_i||_2^2 - ||G - v_i||_2^2\right) + \sum_{t=L_i^{(2)}, L_i^{(3)}}\left(||P - t||_2^2 - ||G - t||_2^2\right) \tag{80}$$

$$= \sum_{i=0}^{k}\left(||P - v_i||_2^2 - ||G - v_i||_2^2\right) + 2(n-k)||P - G||_2^2 \tag{81}$$

The final equation is because both $L_i^{(2)}$ and $L_i^{(3)}$ are in the linear manifold $T_1$ and satisfy Eq. (69). To simplify Eq. (81), we obtain $\sum_{i=0}^{k}\left(||P - v_i||_2^2 - ||G - v_i||_2^2\right) = (k+1)||P - G||_2^2$. Therefore, Theorem 5 (2) holds.

**For Theorem 5 (3).** When $k \geq n$, from Proposition 8, we know that $\mathbf{rank}(v_1 - v_0, v_2 - v_0, ..., v_k - v_0) = n$ holds with probability 1. Hence, if we use the similar deduction from Theorem 5 (1), we can find that Theorem 5 (3) holds when $k \geq n$. On the other hand, when $k < n$, we can get the same result also according to Proposition 8. The reason is that $(v_1 - v_0, v_2 - v_0, ..., v_k - v_0)$ are linearly independent with probability 1.

□

## L   THE GEOMETRIC STRUCTURE OF CONVOLUTIONAL FILTERS.

**Theorem 6.** *Let $v_i \in \mathbb{R}^k$ and $v_i \sim N(\mathbf{0}, c^2 \cdot I_k)$. If $k \to \infty$ and $c \neq 0$, then*

*(1) $||v_i||_2 \approx ||v_j||_2 \to \sqrt{2}c \cdot \frac{\Gamma((k+1)/2)}{\Gamma(k/2)}, 1 \leq i < j \leq N$;*

*(2) $\mathbf{angle}(v_i, v_j) \to \frac{\pi}{2}, 1 \leq i < j \leq N$;*

*(3) $||v_i - v_j||_2 \approx ||v_i - v_t||_2, 1 \leq i < j < t \leq N$;*

*(4) $\mathbb{E}(||v_i||_1)/\mathbf{Var}(||v_i||_1) \to$ a non-zero constant.*

*Proof.* First, since Chebyshev inequality, for $1 \leq i \leq N$ and a given $M$, we have

$$P\left\{|||v_i||_2 - \mathbb{E}(||v_i||_2)| \geq \sqrt{M\mathbf{Var}(||v_i||_2)}\right\} \leq \frac{1}{M}. \tag{82}$$

from Eq. (9), Eq. (10) and Lemma. (2), we can rewrite Eq. (82) when $k \to \infty$:

$$P\left\{||v_i||_2 \in \left[\sqrt{2}c \cdot \frac{\Gamma((k+1)/2)}{\Gamma(k/2)} - \sqrt{\frac{M}{2}}c, \sqrt{2}c \cdot \frac{\Gamma((k+1)/2)}{\Gamma(k/2)} + \sqrt{\frac{M}{2}}c\right]\right\} \geq 1 - \frac{1}{M}. \tag{83}$$

For a small enough $\epsilon > 0$, let $M = 1/\epsilon$. Note that $\sqrt{\frac{M}{2}}c = c/\sqrt{2\epsilon}$ is a constant. When $k \to \infty$, $\sqrt{2}c \cdot \frac{\Gamma((k+1)/2)}{\Gamma(k/2)} \gg \sqrt{\frac{M}{2}}c$. Hence, for any $i \in [1, N]$ and any small enough $\epsilon$, we have

$$P\left\{||v_i||_2 \approx \sqrt{2}c \cdot \frac{\Gamma((k+1)/2)}{\Gamma(k/2)}\right\} \geq 1 - \epsilon. \tag{84}$$

So Theorem 6(1) holds.

Let $v_i = (v_{i1}, v_{i2}, ..., v_{ik})$ and $v_j = (v_{j1}, v_{j2}, ..., v_{jk})$. So $< v_i, v_j > = \sum_{p=1}^k v_{ip}v_{jp}$. Note that, $v_i$ and $v_j$ are independent, hence

$$\mathbb{E}(v_{ip}v_{jp}) = 0, \tag{85}$$

$$\mathbf{Var}(v_{ip}v_{jp}) = \mathbf{Var}(v_{ip})\mathbf{Var}(v_{jp}) + (\mathbb{E}(v_{ip}))^2\mathbf{Var}(v_{jp}) + (\mathbb{E}(v_{jp}))^2\mathbf{Var}(v_{ip}) = 1, \tag{86}$$

since central limit theorem, we have

$$\sqrt{k}\left(\frac{1}{k}\sum_{p=1}^k v_{ip}v_{jp} - 0\right) \sim N(0, 1), \tag{87}$$

According to Eq. (9), Lemma 2 and Eq. (87), when $k \to \infty$, we have

$$\frac{< v_i, v_j >}{||v_i||_2 \cdot ||v_j||_2} \to \frac{1}{\sqrt{k}} \cdot \frac{< v_i, v_j >}{\sqrt{k}} \sim N(0, \frac{1}{k}) \to N(0, 0). \tag{88}$$

So Theorem 6(2) holds. From Theorem 6(1) and Theorem 6(2), Theorem 6(3) can be proved through Pythagoras theorem.

For 6(4), from Proposition. 2, we have

$$\frac{\mathbb{E}(||v_i||_1)}{\mathbf{Var}(||v_i||_1)} = \frac{k \cdot c\sqrt{\frac{2}{\pi}}}{k \cdot c^2(1 - \frac{2}{\pi})} = \frac{\sqrt{\frac{2}{\pi}}}{c(1 - \frac{2}{\pi})} \tag{89}$$

$\square$

As shown in Fig. 21, Theorem 6 actually reveals the geometric structure formed by the parameters of the convolutional filters in CNNs. Specifically, from Theorem 6 (1), the convolutional filters $v_i$ of each layer locate approximately on the surface of $k$ dimensional sphere with $\mathbf{0}$ as the origin and $\sqrt{2}c \cdot \frac{\Gamma((k+1)/2)}{\Gamma(k/2)}$ as the radius. Then, from Theorem 6 (2), the vectors formed by any two different

convolutional filters in the same layer are approximately orthogonal. As this result, for any three different filters $v_1, v_2$ and $v_j$, we can use Pythagoras theorem and Theorem 6 (1) to prove that they are equidistant,*i.e.*, $||v_1 - v_2||_2 \approx ||v_2 - v_3||_2 \approx ||v_3 - v_1||_2$. In fact, Fig. 20 provides another view of the geometric structure of convolutional filters. Since CWDA, $\mathbb{E}(v_i) = \mathbf{0}$. So the correlation matrix $\{(\mathbf{Cor}(v_i, v_j)\}_{N \times N} = c \cdot \{(v_i^T v_j\}_{N \times N}$, where $c$ is a constant. That is to say, there is only one coefficient difference between correlation matrix and Gram matrix. Therefore, the diagonal elements of the matrix are $||v_i||_2^2$, and the off-diagonal elements are the dot product between the convolutional filters. This numerical visualization also verifies the conclusion of Theorem6.

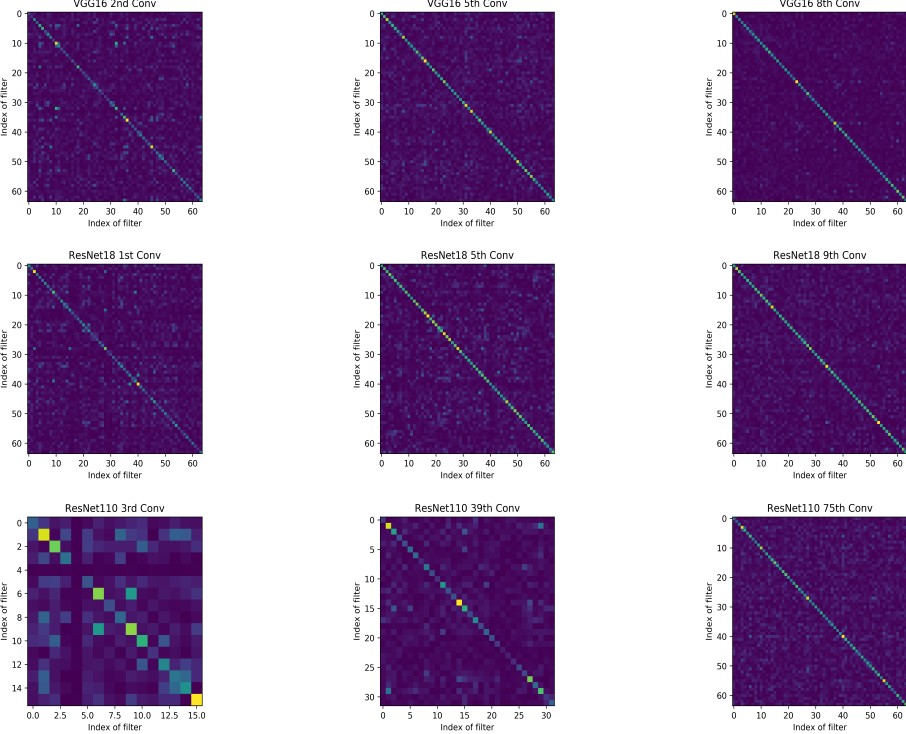

Figure 20: The correlation matrix of convolutional filters. For clarity, we use the first 64 filters in each layer to calculate the Gram matrix.

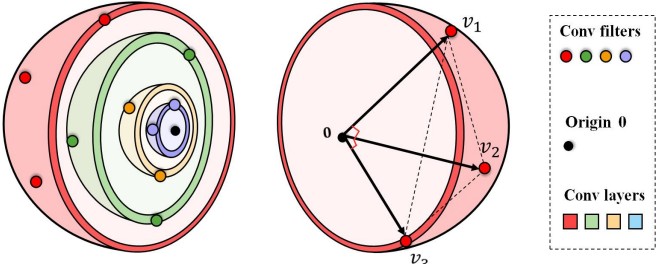

Figure 21: The geometric structure of convolutional filters when the network has large number of filters in each layer. for every pair of filters in one layer, (1) Their $\ell_2$ norm are equivalent ($||v_1||_2 \approx ||v_2||_2 \approx ||v_3||_2$ ); (2) They are equidistant ($||v_1 - v_2||_2 \approx ||v_2 - v_3||_2 \approx ||v_3 - v_1||_2$); (3) and they are orthogonal ($v_1^T v_2 \approx v_2^T v_3 \approx v_3^T v_1 \approx 0$).

# M    THE DETAILS OF OTHER PRUNING CRITERIA

For notation, we denote $i^{\text{th}}$ convolutional filter in layer $l$ as $F_i^l$ and the input feature maps in layer $l$ as $\mathbf{I}^l \in \mathbb{R}^{N \times I^l \times H^l \times W^l}$, where $N, I^l, H^l, W_l$ mean the train set size, number of channels, height and width respectively, $i = 1, 2, \cdots, \lambda_l$, and $l = 1, 2, \cdots, L$. The formulation of the filters' *importance* under each pruning criteria are illustrated as follows:

**Norm-based criteria:**

- $\ell_1$-Norm Li et al. (2016): $||F_i^l||_1$;
- $\ell_2$-Norm Frankle & Carbin (2019): $||F_i^l||_2$;

**BN-based criteria Liu et al. (2017b):**

- BN_$\gamma$: $|\gamma_i^l|$, where $\gamma_i^l$ is the scaling factor in the Batch Normalization layer $l$;
- BN_$\beta$: $|\beta_i^l|$, where $\beta_i^l$ is the shifting factor in the Batch Normalization layer $l$.

**Activation-based criteria:**

- APoZ Hu et al. (2016): $\frac{\sum_{p,q} \mathbb{1}\left(\left(|\mathbf{I}^l * F_i^l|\right)_{p,q} > \sigma\right)}{N \times I^l \times H^l \times W^l}$, where we set $\sigma = 0.0001$ same as Luo & Wu (2017), and $\mathbb{1}(\cdot)$ is the indicator function, $*$ is convolution operator and $\mathbf{I}^l * F_i^l$ is the $i$-th output feature map;
- Entropy Luo & Wu (2017): we first prepare $\mathbf{G}_i^l = GAP(\mathbf{I}^l * F_i^l)$, where $\mathbf{G}_i^l \in \mathbb{R}^{N \times 1}$ and $GAP(\cdot)$ is the Global Average Pooling. Then, we estimate statistical distribution for $\mathbf{G}_i^l$ by dividing all elements in $\mathbf{G}_i^l$ into $m$ bins. Let $p_j$ is the probability of bin $j$, and the the *importance* score is $-\sum_{j=1}^m p_j \log p_j$.

**First order Taylor based criteria Molchanov et al. (2016; 2019a;b):**

- Taylor $\ell_1$-Norm: $||\frac{\partial loss}{\partial F_i^l} \cdot F_i^l||_1$;
- Taylor $\ell_2$-Norm: $||\frac{\partial loss}{\partial F_i^l} \cdot F_i^l||_2$;

The $loss$ is the Cross Entropy Loss on the split training set from the original training set.

# N ADDITIONAL EXPERIMENTS ABOUT IMAGE CLASIFICATION

Table 6: The accuracy(%) of several networks and datasets using different pruning criteria.

| | | Experiment (1) | | | Experiment (2) | | | Experiment (3) | | |
| --- | --- | --- | --- | --- | --- | --- | --- | --- | --- | --- |
| | | Trained | Pruned | Fine-tuned | Trained | Pruned | Fine-tuned | Trained | Pruned | Fine-tuned |
| CIFAR10 | $\ell_1$ | 93.61 | 61.21 | 93.51 | 93.21 | 54.31 | 93.22 | 93.26 | 57.74 | 93.32 |
| VGG16 | $\ell_2$ | 93.61 | 63.41 | 93.32 | 93.21 | 54.61 | 93.42 | 93.26 | 57.42 | 93.29 |
| | **GM** | 93.61 | 61.22 | 93.41 | 93.21 | 53.71 | 93.25 | 93.26 | 57.46 | 93.36 |
| CIFAR100 | $\ell_1$ | 72.67 | 25.91 | 71.50 | 72.99 | 20.43 | 71.36 | 72.56 | 24.01 | 71.07 |
| VGG16 | $\ell_2$ | 72.67 | 27.07 | 71.28 | 72.99 | 22.31 | 71.12 | 72.56 | 24.45 | 70.92 |
| | **GM** | 72.67 | 26.37 | 71.27 | 72.99 | 21.67 | 71.26 | 72.56 | 24.26 | 70.78 |
| ImageNet | $\ell_1$ | 71.58 | 30.33 | 71.02 | 71.33 | 40.33 | 70.12 | 72.01 | 28.07 | 70.93 |
| VGG16 | $\ell_2$ | 71.58 | 29.47 | 70.83 | 71.33 | 40.45 | 70.13 | 72.01 | 27.89 | 71.02 |
| | **GM** | 71.58 | 30.76 | 70.95 | 71.33 | 39.86 | 70.33 | 72.01 | 28.01 | 70.74 |
| CIFAR10 | $\ell_1$ | 92.98 | 77.73 | 93.08 | 92.97 | 76.02 | 92.82 | 93.01 | 79.93 | 92.81 |
| ResNet56 | $\ell_2$ | 92.98 | 79.02 | 92.83 | 92.97 | 77.91 | 92.72 | 93.01 | 82.43 | 92.81 |
| | **GM** | 92.98 | 74.26 | 92.77 | 93.2 | 73.93 | 92.61 | 93.01 | 80.48 | 92.84 |
| CIFAR100 | $\ell_1$ | 71.36 | 50.64 | 70.15 | 70.02 | 52.41 | 69.19 | 70.48 | 52.19 | 69.77 |
| ResNet56 | $\ell_2$ | 71.36 | 53.44 | 70.16 | 70.02 | 52.73 | 69.31 | 70.48 | 52.16 | 69.62 |
| | **GM** | 71.36 | 45.12 | 70.22 | 70.02 | 52.62 | 69.54 | 70.48 | 50.74 | 69.69 |
| ImageNet | $\ell_1$ | 73.31 | 62.22 | 73.06 | 73.16 | 54.24 | 72.99 | 73.21 | 63.12 | 73.02 |
| ResNet34 | $\ell_2$ | 73.31 | 62.02 | 72.91 | 73.16 | 53.64 | 72.78 | 73.21 | 62.98 | 72.86 |
| | **GM** | 73.31 | 61.88 | 72.96 | 73.16 | 53.48 | 72.94 | 73.21 | 62.36 | 73.04 |

All the setting of these experiments are under can be found in `https://github.com/bearpaw/pytorch-classification`. Specifically, for pruning ratio:

VGG16 on CIFAR10, CIFAR100 and ImageNet:

```
https://github.com/Eric-mingjie/rethinking-network-pruning/blob/
master/cifar/l1-norm-pruning/vggprune.py#L84
```

ResNet56 on CIFAR10 and CIFAR100:

```
https://github.com/Eric-mingjie/rethinking-network-pruning/blob/
master/cifar/l1-norm-pruning/res56prune.py#L94
```

ResNet34 on ImageNet:

```
https://github.com/Eric-mingjie/rethinking-network-pruning/blob/
master/imagenet/l1-norm-pruning/prune.py#L138
```

# O ABOUT WEIGHT DECAY

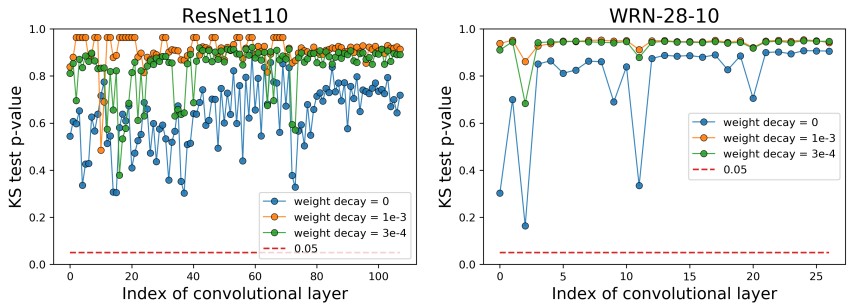

Figure 22: KS test (Lilliefors, 1967) while using different settings of weight decay.

We train the ResNet110 and WRN-28-10 on CIFAR100 with different weight decay (1e-3, 3e-4 and 0) and use KS test to verify whether the parameters of different layers follow a normal distribution. In Fig. 22, we can find

(1) When weight decay (wd) is non-zero, the normality is higher than that when weight decay is 0.

(2) If weight decay is 0, the p-value can still be much greater than 0.05, which means that the regularization of weight decay may not be the key reason for CWDA. The distribution of the parameters in these two networks (weight decay is 0) are shown in Fig. 24 and Fig. 23.

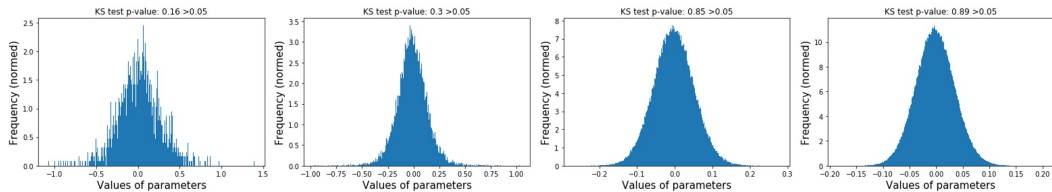

Figure 23: The distribution of parameters in different convolutional filters (WRN-28-10, wd = 0).

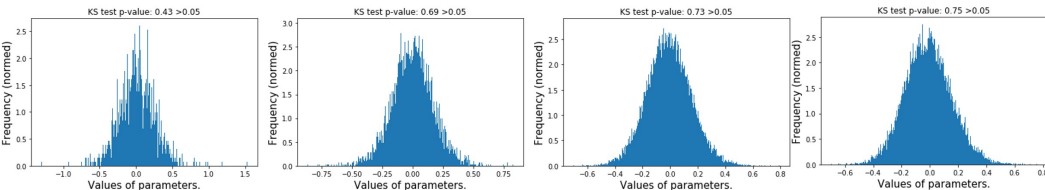

Figure 24: The distribution of parameters in different convolutional filters (ResNet110, wd = 0).

# P    STATISTICAL TEST

In this section, according to Table 3 and Section 2.1, we have a series of statistical tests for the necessary conditions of CWDA. let $F_{ij} \in \mathbb{R}^{N_i \times k \times k}$ represent the $j^{\text{th}}$ filter of the $i^{\text{th}}$ convolutional layer.

(1) **Gaussian** (*i.e.*, to verify whether $F_{ij}$ approximatively follow a Gaussian-alike distribution.). In $i^{\text{th}}$ layer, we use Kolmogorov–Smirnov (KS) test (Lilliefors, 1967) to check if all the weights in the same layer follow a normal distribution.

(2) **Standard Deviation** (*i.e.*, to verify whether the standard deviation of each filter in any layers tends to be a constant $c$.). Let $\sigma_j$ denotes the standard deviation of all the weights of filter $F_{ij}$ in $i^{\text{th}}$ layer. We use Student's t test (Efron, 1969) to check if the variance of these $\sigma_j$ is small enough. The null hypothesis $H_0$ and the alternative hypothesis $H_1$ are:

$$H_0 : \mathbf{Var}(\sigma_1, \sigma_2, .., \sigma_{N_i}) \le \sigma_0^2, \qquad H_1 : \mathbf{Var}(\sigma_1, \sigma_2, .., \sigma_{N_i}) > \sigma_0^2.$$

where $N_i$ denotes the number of the filters in $i^{\text{th}}$ layer and $\sigma_0$ is a given real number which is small enough, like $\sigma_0^2 = 0.0001$.

(3) **Mean** (*i.e.*, to verify whether the mean of $F_{ij}$ is 0.). Let the mean of all the weights in the same layer is $\mu$. We use Student's t test (Efron, 1969) to check if $\mu$ is close to 0. First, we check the upper bound (Mean-Left) of $\mu$, *i.e.*,

$$H_0 : \mu \le \epsilon, \qquad H_1 : \mu > \epsilon.$$

where $\epsilon$ is a small constant, like $\epsilon = 0.01$. Next, we check the lower bound (Mean-Right) and the null hypothesis $H_0$ and the alternative hypothesis $H_1$ are:

$$H_0 : \mu \ge -\epsilon, \qquad H_1 : \mu < -\epsilon.$$

Of course, the $p$ value for $H_0 : \mu \ge -\epsilon$ and $H_0 : \mu \le \epsilon$ should be the same. There are several Notable points:

- In all the statistical tests, let the confidence level be 0.95, $\epsilon = 0.01$ and $\sigma_0^2 = 0.0001$.

- we use Green color to represent that the convolutional filters in one layer can pass the statistical test. Conversely, the Red color means that the filters can not pass the tests.

- In most layers, the convolutional filters can pass the statistical tests, except for a few layers which are in front of the network. This phenomenon is consistent with the analysis in Section 5.1 and it does not mean CWDA is not true.

- Most of the experiments are image classification, except for the tests in Appendix P.7.

- **p-value** and **c-value** denote $p$ value and critical value (confidence level is 0.95), respectively. **t-value** is $t$ values in Student's t test. If **p-value** is larger than **c-value** or **t-value** is smaller than **c-value**, we think this filter passes the tests.

- In fact, some of the experiments in Table 3 are repeated. These experiments are shown in following tables. The (*) means that ,for the sake of brevity, the repeat experiments are shown only once on the experiment with (*).

Table 7: The repeated experiments in Table 3.

| | | **NETWORK STRUCTURE** (P.1) | **OPTIMIZER** (P.2) | **INITIALIZATION** (P.5) |
|---|---|---|---|---|
| (1) | | ResNet (*) | SGD | kaiming-normal |
| (2) | | **NETWORK STRUCTURE** (P.1) | **REGULARIZATION** (P.3) | |
| | | WRN (*) | L2 norm | |
| (3) | | **BATCH NORMALIZATION** (P.8) | **LEARNING RATE** (P.10) | |
| | | VGG-bn (*) | Schedule 150-225 | |

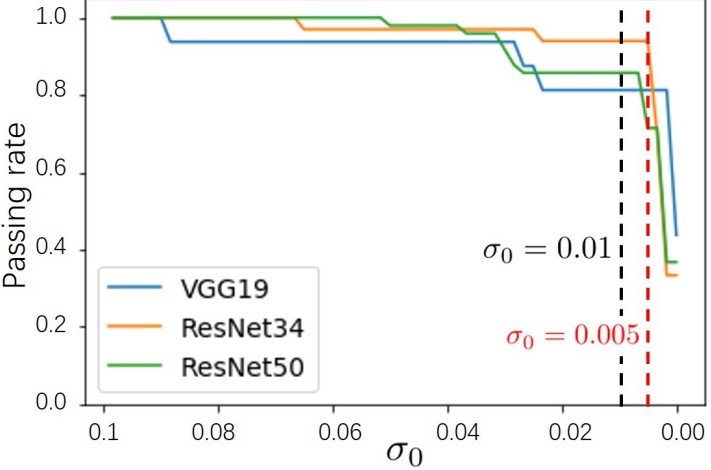

Figure 25: The passing rate of statistical test in Appendix P(2), where $0 < \sigma_0 \leq 0.1$ and passing rate is the ratio between the number of the filters passed the test and the number of total filters.

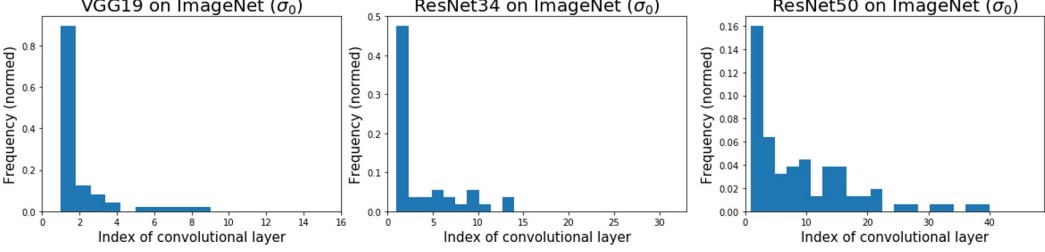

Figure 26: The filters that can not pass the statistical test in Appendix P(2). We record the convolutional filters that can not pass the statistical test under all settings of $\sigma_0$ in Fig. 25. It can be found that the filters that fail to pass the test are concentrated in the first few layers. This is consistent with the statement in Section 5.1.

## P.1 NETWORK STRUCTURE

config: `https://github.com/bearpaw/pytorch-classification`.

Table 8: Cifar100 ResNet164

| Layer | Number | dim | Gaussian | | Mean_Left | | Mean_Right | | Sigma | |
|---|---|---|---|---|---|---|---|---|---|---|
| | | | p-value | c-value | p-value | c-value | p-value | c-value | t-value | c-value |
| Conv1 | 16 | 16 | 0.01 | 0.05 | 0.56 | 0.05 | 0.56 | 0.05 | 448.68 | 26.3 |
| Conv2 | 16 | 144 | 0.04 | 0.05 | 0.68 | 0.05 | 0.68 | 0.05 | 114.12 | 26.3 |
| Conv3 | 64 | 16 | 0.08 | 0.05 | 0.63 | 0.05 | 0.63 | 0.05 | 661.15 | 83.68 |
| Conv4 | 16 | 64 | 0.44 | 0.05 | 0.63 | 0.05 | 0.63 | 0.05 | 17.16 | 26.3 |
| Conv5 | 16 | 144 | 0.31 | 0.05 | 0.68 | 0.05 | 0.68 | 0.05 | 21.0 | 26.3 |
| Conv6 | 64 | 16 | 0.36 | 0.05 | 0.63 | 0.05 | 0.63 | 0.05 | 112.55 | 83.68 |
| Conv7 | 16 | 64 | 0.58 | 0.05 | 0.63 | 0.05 | 0.63 | 0.05 | 4.73 | 26.3 |
| Conv8 | 16 | 144 | 0.35 | 0.05 | 0.68 | 0.05 | 0.68 | 0.05 | 11.9 | 26.3 |
| Conv9 | 64 | 16 | 0.38 | 0.05 | 0.63 | 0.05 | 0.63 | 0.05 | 57.19 | 83.68 |
| Conv10 | 16 | 64 | 0.69 | 0.05 | 0.63 | 0.05 | 0.63 | 0.05 | 2.61 | 26.3 |
| Conv11 | 16 | 144 | 0.57 | 0.05 | 0.68 | 0.05 | 0.68 | 0.05 | 5.19 | 26.3 |
| Conv12 | 64 | 16 | 0.52 | 0.05 | 0.63 | 0.05 | 0.63 | 0.05 | 41.82 | 83.68 |
| Conv13 | 16 | 64 | 0.71 | 0.05 | 0.63 | 0.05 | 0.63 | 0.05 | 1.4 | 26.3 |
| Conv14 | 16 | 144 | 0.62 | 0.05 | 0.68 | 0.05 | 0.68 | 0.05 | 2.9 | 26.3 |
| Conv15 | 64 | 16 | 0.64 | 0.05 | 0.63 | 0.05 | 0.63 | 0.05 | 18.71 | 83.68 |
| Conv16 | 16 | 64 | 0.83 | 0.05 | 0.63 | 0.05 | 0.63 | 0.05 | 1.35 | 26.3 |
| Conv17 | 16 | 144 | 0.69 | 0.05 | 0.68 | 0.05 | 0.68 | 0.05 | 1.27 | 26.3 |
| Conv18 | 64 | 16 | 0.84 | 0.05 | 0.63 | 0.05 | 0.63 | 0.05 | 9.65 | 83.68 |
| Conv19 | 16 | 64 | 0.66 | 0.05 | 0.63 | 0.05 | 0.63 | 0.05 | 2.89 | 26.3 |
| Conv20 | 16 | 144 | 0.7 | 0.05 | 0.68 | 0.05 | 0.68 | 0.05 | 0.77 | 26.3 |
| Conv21 | 64 | 16 | 0.82 | 0.05 | 0.63 | 0.05 | 0.63 | 0.05 | 8.08 | 83.68 |
| Conv22 | 16 | 64 | 0.81 | 0.05 | 0.63 | 0.05 | 0.63 | 0.05 | 1.8 | 26.3 |
| Conv23 | 16 | 144 | 0.88 | 0.05 | 0.68 | 0.05 | 0.68 | 0.05 | 0.51 | 26.3 |
| Conv24 | 64 | 16 | 0.86 | 0.05 | 0.63 | 0.05 | 0.63 | 0.05 | 4.45 | 83.68 |
| Conv25 | 16 | 64 | 0.87 | 0.05 | 0.63 | 0.05 | 0.63 | 0.05 | 1.24 | 26.3 |
| Conv26 | 16 | 144 | 0.84 | 0.05 | 0.68 | 0.05 | 0.68 | 0.05 | 0.68 | 26.3 |
| Conv27 | 64 | 16 | 0.87 | 0.05 | 0.63 | 0.05 | 0.63 | 0.05 | 4.07 | 83.68 |
| Conv28 | 16 | 64 | 0.47 | 0.05 | 0.63 | 0.05 | 0.63 | 0.05 | 4.45 | 26.3 |
| Conv29 | 16 | 144 | 0.64 | 0.05 | 0.68 | 0.05 | 0.68 | 0.05 | 4.66 | 26.3 |
| Conv30 | 64 | 16 | 0.8 | 0.05 | 0.63 | 0.05 | 0.63 | 0.05 | 12.15 | 83.68 |
| Conv31 | 16 | 64 | 0.69 | 0.05 | 0.63 | 0.05 | 0.63 | 0.05 | 2.59 | 26.3 |
| Conv32 | 16 | 144 | 0.84 | 0.05 | 0.68 | 0.05 | 0.68 | 0.05 | 1.03 | 26.3 |
| Conv33 | 64 | 16 | 0.83 | 0.05 | 0.63 | 0.05 | 0.63 | 0.05 | 10.12 | 83.68 |
| Conv34 | 16 | 64 | 0.52 | 0.05 | 0.63 | 0.05 | 0.63 | 0.05 | 6.17 | 26.3 |
| Conv35 | 16 | 144 | 0.38 | 0.05 | 0.68 | 0.05 | 0.68 | 0.05 | 6.31 | 26.3 |
| Conv36 | 64 | 16 | 0.67 | 0.05 | 0.63 | 0.05 | 0.63 | 0.05 | 12.34 | 83.68 |
| Conv37 | 16 | 64 | 0.66 | 0.05 | 0.63 | 0.05 | 0.63 | 0.05 | 5.25 | 26.3 |
| Conv38 | 16 | 144 | 0.69 | 0.05 | 0.68 | 0.05 | 0.68 | 0.05 | 2.07 | 26.3 |
| Conv39 | 64 | 16 | 0.84 | 0.05 | 0.63 | 0.05 | 0.63 | 0.05 | 7.37 | 83.68 |
| Conv40 | 16 | 64 | 0.63 | 0.05 | 0.63 | 0.05 | 0.63 | 0.05 | 6.93 | 26.3 |
| Conv41 | 16 | 144 | 0.52 | 0.05 | 0.68 | 0.05 | 0.68 | 0.05 | 4.62 | 26.3 |
| Conv42 | 64 | 16 | 0.69 | 0.05 | 0.63 | 0.05 | 0.63 | 0.05 | 18.53 | 83.68 |
| Conv43 | 16 | 64 | 0.64 | 0.05 | 0.63 | 0.05 | 0.63 | 0.05 | 5.38 | 26.3 |
| Conv44 | 16 | 144 | 0.6 | 0.05 | 0.68 | 0.05 | 0.68 | 0.05 | 2.7 | 26.3 |
| Conv45 | 64 | 16 | 0.69 | 0.05 | 0.63 | 0.05 | 0.63 | 0.05 | 17.9 | 83.68 |
| Conv46 | 16 | 64 | 0.82 | 0.05 | 0.63 | 0.05 | 0.63 | 0.05 | 2.32 | 26.3 |
| Conv47 | 16 | 144 | 0.82 | 0.05 | 0.68 | 0.05 | 0.68 | 0.05 | 1.98 | 26.3 |
| Conv48 | 64 | 16 | 0.82 | 0.05 | 0.63 | 0.05 | 0.63 | 0.05 | 9.34 | 83.68 |
| Conv49 | 16 | 64 | 0.62 | 0.05 | 0.63 | 0.05 | 0.63 | 0.05 | 5.16 | 26.3 |

| | | | | | | | | | | | |
|---|---|---|---|---|---|---|---|---|---|---|---|
| Conv50 | 16 | 144 | 0.39 | 0.05 | 0.68 | 0.05 | 0.68 | 0.05 | 6.45 | 26.3 |
| Conv51 | 64 | 16 | 0.82 | 0.05 | 0.63 | 0.05 | 0.63 | 0.05 | 14.56 | 83.68 |
| Conv52 | 16 | 64 | 0.54 | 0.05 | 0.63 | 0.05 | 0.63 | 0.05 | 6.05 | 26.3 |
| Conv53 | 16 | 144 | 0.81 | 0.05 | 0.68 | 0.05 | 0.68 | 0.05 | 4.12 | 26.3 |
| Conv54 | 64 | 16 | 0.67 | 0.05 | 0.63 | 0.05 | 0.63 | 0.05 | 15.76 | 83.68 |
| Conv55 | 32 | 64 | 0.31 | 0.05 | 0.67 | 0.05 | 0.67 | 0.05 | 26.33 | 46.19 |
| Conv56 | 32 | 288 | 0.56 | 0.05 | 0.83 | 0.05 | 0.83 | 0.05 | 6.36 | 46.19 |
| Conv57 | 128 | 32 | 0.47 | 0.05 | 0.74 | 0.05 | 0.74 | 0.05 | 94.73 | 155.4 |
| Conv58 | 32 | 128 | 0.56 | 0.05 | 0.74 | 0.05 | 0.74 | 0.05 | 7.21 | 46.19 |
| Conv59 | 32 | 288 | 0.52 | 0.05 | 0.83 | 0.05 | 0.83 | 0.05 | 2.94 | 46.19 |
| Conv60 | 128 | 32 | 0.63 | 0.05 | 0.74 | 0.05 | 0.74 | 0.05 | 27.0 | 155.4 |
| Conv61 | 32 | 128 | 0.69 | 0.05 | 0.74 | 0.05 | 0.74 | 0.05 | 1.27 | 46.19 |
| Conv62 | 32 | 288 | 0.84 | 0.05 | 0.83 | 0.05 | 0.83 | 0.05 | 1.94 | 46.19 |
| Conv63 | 128 | 32 | 0.83 | 0.05 | 0.74 | 0.05 | 0.74 | 0.05 | 17.24 | 155.4 |
| Conv64 | 32 | 128 | 0.31 | 0.05 | 0.74 | 0.05 | 0.74 | 0.05 | 7.84 | 46.19 |
| Conv65 | 32 | 288 | 0.41 | 0.05 | 0.83 | 0.05 | 0.83 | 0.05 | 4.86 | 46.19 |
| Conv66 | 128 | 32 | 0.81 | 0.05 | 0.74 | 0.05 | 0.74 | 0.05 | 33.53 | 155.4 |
| Conv67 | 32 | 128 | 0.51 | 0.05 | 0.74 | 0.05 | 0.74 | 0.05 | 5.16 | 46.19 |
| Conv68 | 32 | 288 | 0.83 | 0.05 | 0.83 | 0.05 | 0.83 | 0.05 | 2.86 | 46.19 |
| Conv69 | 128 | 32 | 0.68 | 0.05 | 0.74 | 0.05 | 0.74 | 0.05 | 29.78 | 155.4 |
| Conv70 | 32 | 128 | 0.83 | 0.05 | 0.74 | 0.05 | 0.74 | 0.05 | 1.27 | 46.19 |
| Conv71 | 32 | 288 | 0.69 | 0.05 | 0.83 | 0.05 | 0.83 | 0.05 | 1.87 | 46.19 |
| Conv72 | 128 | 32 | 0.84 | 0.05 | 0.74 | 0.05 | 0.74 | 0.05 | 9.81 | 155.4 |
| Conv73 | 32 | 128 | 0.66 | 0.05 | 0.74 | 0.05 | 0.74 | 0.05 | 1.89 | 46.19 |
| Conv74 | 32 | 288 | 0.83 | 0.05 | 0.83 | 0.05 | 0.83 | 0.05 | 1.06 | 46.19 |
| Conv75 | 128 | 32 | 0.87 | 0.05 | 0.74 | 0.05 | 0.74 | 0.05 | 13.07 | 155.4 |
| Conv76 | 32 | 128 | 0.83 | 0.05 | 0.74 | 0.05 | 0.74 | 0.05 | 1.93 | 46.19 |
| Conv77 | 32 | 288 | 0.86 | 0.05 | 0.83 | 0.05 | 0.83 | 0.05 | 1.74 | 46.19 |
| Conv78 | 128 | 32 | 0.86 | 0.05 | 0.74 | 0.05 | 0.74 | 0.05 | 12.89 | 155.4 |
| Conv79 | 32 | 128 | 0.46 | 0.05 | 0.74 | 0.05 | 0.74 | 0.05 | 5.41 | 46.19 |
| Conv80 | 32 | 288 | 0.84 | 0.05 | 0.83 | 0.05 | 0.83 | 0.05 | 1.63 | 46.19 |
| Conv81 | 128 | 32 | 0.7 | 0.05 | 0.74 | 0.05 | 0.74 | 0.05 | 22.23 | 155.4 |
| Conv82 | 32 | 128 | 0.57 | 0.05 | 0.74 | 0.05 | 0.74 | 0.05 | 4.95 | 46.19 |
| Conv83 | 32 | 288 | 0.69 | 0.05 | 0.83 | 0.05 | 0.83 | 0.05 | 2.5 | 46.19 |
| Conv84 | 128 | 32 | 0.8 | 0.05 | 0.74 | 0.05 | 0.74 | 0.05 | 24.7 | 155.4 |
| Conv85 | 32 | 128 | 0.82 | 0.05 | 0.74 | 0.05 | 0.74 | 0.05 | 1.88 | 46.19 |
| Conv86 | 32 | 288 | 0.81 | 0.05 | 0.83 | 0.05 | 0.83 | 0.05 | 1.88 | 46.19 |
| Conv87 | 128 | 32 | 0.82 | 0.05 | 0.74 | 0.05 | 0.74 | 0.05 | 19.06 | 155.4 |
| Conv88 | 32 | 128 | 0.8 | 0.05 | 0.74 | 0.05 | 0.74 | 0.05 | 2.68 | 46.19 |
| Conv89 | 32 | 288 | 0.69 | 0.05 | 0.83 | 0.05 | 0.83 | 0.05 | 2.5 | 46.19 |
| Conv90 | 128 | 32 | 0.69 | 0.05 | 0.74 | 0.05 | 0.74 | 0.05 | 15.94 | 155.4 |
| Conv91 | 32 | 128 | 0.68 | 0.05 | 0.74 | 0.05 | 0.74 | 0.05 | 3.88 | 46.19 |
| Conv92 | 32 | 288 | 0.82 | 0.05 | 0.83 | 0.05 | 0.83 | 0.05 | 2.55 | 46.19 |
| Conv93 | 128 | 32 | 0.66 | 0.05 | 0.74 | 0.05 | 0.74 | 0.05 | 19.66 | 155.4 |
| Conv94 | 32 | 128 | 0.52 | 0.05 | 0.74 | 0.05 | 0.74 | 0.05 | 6.16 | 46.19 |
| Conv95 | 32 | 288 | 0.6 | 0.05 | 0.83 | 0.05 | 0.83 | 0.05 | 2.52 | 46.19 |
| Conv96 | 128 | 32 | 0.81 | 0.05 | 0.74 | 0.05 | 0.74 | 0.05 | 22.61 | 155.4 |
| Conv97 | 32 | 128 | 0.83 | 0.05 | 0.74 | 0.05 | 0.74 | 0.05 | 3.05 | 46.19 |
| Conv98 | 32 | 288 | 0.62 | 0.05 | 0.83 | 0.05 | 0.83 | 0.05 | 2.4 | 46.19 |
| Conv99 | 128 | 32 | 0.7 | 0.05 | 0.74 | 0.05 | 0.74 | 0.05 | 14.79 | 155.4 |
| Conv100 | 32 | 128 | 0.32 | 0.05 | 0.74 | 0.05 | 0.74 | 0.05 | 6.89 | 46.19 |
| Conv101 | 32 | 288 | 0.57 | 0.05 | 0.83 | 0.05 | 0.83 | 0.05 | 3.2 | 46.19 |
| Conv102 | 128 | 32 | 0.65 | 0.05 | 0.74 | 0.05 | 0.74 | 0.05 | 20.08 | 155.4 |
| Conv103 | 32 | 128 | 0.65 | 0.05 | 0.74 | 0.05 | 0.74 | 0.05 | 3.7 | 46.19 |
| Conv104 | 32 | 288 | 0.66 | 0.05 | 0.83 | 0.05 | 0.83 | 0.05 | 1.77 | 46.19 |
| Conv105 | 128 | 32 | 0.7 | 0.05 | 0.74 | 0.05 | 0.74 | 0.05 | 20.8 | 155.4 |
| Conv106 | 32 | 128 | 0.55 | 0.05 | 0.74 | 0.05 | 0.74 | 0.05 | 5.14 | 46.19 |
| Conv107 | 32 | 288 | 0.81 | 0.05 | 0.83 | 0.05 | 0.83 | 0.05 | 1.47 | 46.19 |
| Conv108 | 128 | 32 | 0.55 | 0.05 | 0.74 | 0.05 | 0.74 | 0.05 | 21.77 | 155.4 |

| | | | | | | | | | | |
|---|---|---|---|---|---|---|---|---|---|---|
| Conv109 | 64 | 128 | 0.5 | 0.05 | 0.82 | 0.05 | 0.82 | 0.05 | 8.73 | 83.68 |
| Conv110 | 64 | 576 | 0.81 | 0.05 | 0.97 | 0.05 | 0.97 | 0.05 | 1.37 | 83.68 |
| Conv111 | 256 | 64 | 0.47 | 0.05 | 0.9 | 0.05 | 0.9 | 0.05 | 90.92 | 294.32 |
| Conv112 | 64 | 256 | 0.67 | 0.05 | 0.9 | 0.05 | 0.9 | 0.05 | 2.2 | 83.68 |
| Conv113 | 64 | 576 | 0.84 | 0.05 | 0.97 | 0.05 | 0.97 | 0.05 | 1.43 | 83.68 |
| Conv114 | 256 | 64 | 0.66 | 0.05 | 0.9 | 0.05 | 0.9 | 0.05 | 26.24 | 294.32 |
| Conv115 | 64 | 256 | 0.63 | 0.05 | 0.9 | 0.05 | 0.9 | 0.05 | 1.96 | 83.68 |
| Conv116 | 64 | 576 | 0.81 | 0.05 | 0.97 | 0.05 | 0.97 | 0.05 | 1.45 | 83.68 |
| Conv117 | 256 | 64 | 0.63 | 0.05 | 0.9 | 0.05 | 0.9 | 0.05 | 23.12 | 294.32 |
| Conv118 | 64 | 256 | 0.81 | 0.05 | 0.9 | 0.05 | 0.9 | 0.05 | 1.66 | 83.68 |
| Conv119 | 64 | 576 | 0.82 | 0.05 | 0.97 | 0.05 | 0.97 | 0.05 | 1.15 | 83.68 |
| Conv120 | 256 | 64 | 0.8 | 0.05 | 0.9 | 0.05 | 0.9 | 0.05 | 18.65 | 294.32 |
| Conv121 | 64 | 256 | 0.71 | 0.05 | 0.9 | 0.05 | 0.9 | 0.05 | 2.41 | 83.68 |
| Conv122 | 64 | 576 | 0.85 | 0.05 | 0.97 | 0.05 | 0.97 | 0.05 | 1.07 | 83.68 |
| Conv123 | 256 | 64 | 0.66 | 0.05 | 0.9 | 0.05 | 0.9 | 0.05 | 19.45 | 294.32 |
| Conv124 | 64 | 256 | 0.81 | 0.05 | 0.9 | 0.05 | 0.9 | 0.05 | 1.33 | 83.68 |
| Conv125 | 64 | 576 | 0.86 | 0.05 | 0.97 | 0.05 | 0.97 | 0.05 | 0.86 | 83.68 |
| Conv126 | 256 | 64 | 0.81 | 0.05 | 0.9 | 0.05 | 0.9 | 0.05 | 17.83 | 294.32 |
| Conv127 | 64 | 256 | 0.83 | 0.05 | 0.9 | 0.05 | 0.9 | 0.05 | 1.5 | 83.68 |
| Conv128 | 64 | 576 | 0.86 | 0.05 | 0.97 | 0.05 | 0.97 | 0.05 | 1.33 | 83.68 |
| Conv129 | 256 | 64 | 0.81 | 0.05 | 0.9 | 0.05 | 0.9 | 0.05 | 17.73 | 294.32 |
| Conv130 | 64 | 256 | 0.83 | 0.05 | 0.9 | 0.05 | 0.9 | 0.05 | 1.45 | 83.68 |
| Conv131 | 64 | 576 | 0.83 | 0.05 | 0.97 | 0.05 | 0.97 | 0.05 | 0.89 | 83.68 |
| Conv132 | 256 | 64 | 0.68 | 0.05 | 0.9 | 0.05 | 0.9 | 0.05 | 18.82 | 294.32 |
| Conv133 | 64 | 256 | 0.69 | 0.05 | 0.9 | 0.05 | 0.9 | 0.05 | 1.7 | 83.68 |
| Conv134 | 64 | 576 | 0.83 | 0.05 | 0.97 | 0.05 | 0.97 | 0.05 | 1.05 | 83.68 |
| Conv135 | 256 | 64 | 0.66 | 0.05 | 0.9 | 0.05 | 0.9 | 0.05 | 19.59 | 294.32 |
| Conv136 | 64 | 256 | 0.84 | 0.05 | 0.9 | 0.05 | 0.9 | 0.05 | 1.48 | 83.68 |
| Conv137 | 64 | 576 | 0.85 | 0.05 | 0.97 | 0.05 | 0.97 | 0.05 | 0.62 | 83.68 |
| Conv138 | 256 | 64 | 0.67 | 0.05 | 0.9 | 0.05 | 0.9 | 0.05 | 17.88 | 294.32 |
| Conv139 | 64 | 256 | 0.81 | 0.05 | 0.9 | 0.05 | 0.9 | 0.05 | 1.49 | 83.68 |
| Conv140 | 64 | 576 | 0.83 | 0.05 | 0.97 | 0.05 | 0.97 | 0.05 | 0.59 | 83.68 |
| Conv141 | 256 | 64 | 0.82 | 0.05 | 0.9 | 0.05 | 0.9 | 0.05 | 18.42 | 294.32 |
| Conv142 | 64 | 256 | 0.67 | 0.05 | 0.9 | 0.05 | 0.9 | 0.05 | 1.39 | 83.68 |
| Conv143 | 64 | 576 | 0.85 | 0.05 | 0.97 | 0.05 | 0.97 | 0.05 | 0.68 | 83.68 |
| Conv144 | 256 | 64 | 0.8 | 0.05 | 0.9 | 0.05 | 0.9 | 0.05 | 19.86 | 294.32 |
| Conv145 | 64 | 256 | 0.63 | 0.05 | 0.9 | 0.05 | 0.9 | 0.05 | 1.76 | 83.68 |
| Conv146 | 64 | 576 | 0.84 | 0.05 | 0.97 | 0.05 | 0.97 | 0.05 | 0.77 | 83.68 |
| Conv147 | 256 | 64 | 0.81 | 0.05 | 0.9 | 0.05 | 0.9 | 0.05 | 21.98 | 294.32 |
| Conv148 | 64 | 256 | 0.65 | 0.05 | 0.9 | 0.05 | 0.9 | 0.05 | 1.51 | 83.68 |
| Conv149 | 64 | 576 | 0.85 | 0.05 | 0.97 | 0.05 | 0.97 | 0.05 | 0.82 | 83.68 |
| Conv150 | 256 | 64 | 0.65 | 0.05 | 0.9 | 0.05 | 0.9 | 0.05 | 24.23 | 294.32 |
| Conv151 | 64 | 256 | 0.68 | 0.05 | 0.9 | 0.05 | 0.9 | 0.05 | 1.8 | 83.68 |
| Conv152 | 64 | 576 | 0.83 | 0.05 | 0.97 | 0.05 | 0.97 | 0.05 | 1.09 | 83.68 |
| Conv153 | 256 | 64 | 0.63 | 0.05 | 0.9 | 0.05 | 0.9 | 0.05 | 22.33 | 294.32 |
| Conv154 | 64 | 256 | 0.69 | 0.05 | 0.9 | 0.05 | 0.9 | 0.05 | 1.2 | 83.68 |
| Conv155 | 64 | 576 | 0.85 | 0.05 | 0.97 | 0.05 | 0.97 | 0.05 | 1.36 | 83.68 |
| Conv156 | 256 | 64 | 0.53 | 0.05 | 0.9 | 0.05 | 0.9 | 0.05 | 31.15 | 294.32 |
| Conv157 | 64 | 256 | 0.64 | 0.05 | 0.9 | 0.05 | 0.9 | 0.05 | 1.6 | 83.68 |
| Conv158 | 64 | 576 | 0.82 | 0.05 | 0.97 | 0.05 | 0.97 | 0.05 | 1.08 | 83.68 |
| Conv159 | 256 | 64 | 0.49 | 0.05 | 0.9 | 0.05 | 0.9 | 0.05 | 43.56 | 294.32 |
| Conv160 | 64 | 256 | 0.5 | 0.05 | 0.9 | 0.05 | 0.9 | 0.05 | 1.94 | 83.68 |
| Conv161 | 64 | 576 | 0.85 | 0.05 | 0.97 | 0.05 | 0.97 | 0.05 | 0.36 | 83.68 |
| Conv162 | 256 | 64 | 0.36 | 0.05 | 0.9 | 0.05 | 0.9 | 0.05 | 137.02 | 294.32 |
| **Passing rate** | - | - | 98.77% | | 100.0% | | 100.0% | | 97.53% | |

Table 9: Cifar100 VGG19

| Layer | Number | dim | Gaussian | | Mean_Left | | Mean_Right | | Sigma | |
|---|---|---|---|---|---|---|---|---|---|---|
| | | | p-value | c-value | p-value | c-value | p-value | c-value | t-value | c-value |
| Conv1 | 64 | 27 | 0.24 | 0.05 | 0.66 | 0.05 | 0.66 | 0.05 | 396.02 | 83.68 |
| Conv2 | 64 | 576 | 0.51 | 0.05 | 0.97 | 0.05 | 0.97 | 0.05 | 1.73 | 83.68 |
| Conv3 | 128 | 576 | 0.82 | 0.05 | 1.00 | 0.05 | 1.00 | 0.05 | 2.7 | 155.4 |
| Conv4 | 128 | 1152 | 0.84 | 0.05 | 1.00 | 0.05 | 1.00 | 0.05 | 0.37 | 155.4 |
| Conv5 | 256 | 1152 | 0.86 | 0.05 | 1.00 | 0.05 | 1.00 | 0.05 | 1.38 | 294.32 |
| Conv6 | 256 | 2304 | 0.88 | 0.05 | 1.00 | 0.05 | 1.00 | 0.05 | 0.6 | 294.32 |
| Conv7 | 256 | 2304 | 0.91 | 0.05 | 1.00 | 0.05 | 1.00 | 0.05 | 0.38 | 294.32 |
| Conv8 | 256 | 2304 | 0.91 | 0.05 | 1.00 | 0.05 | 1.00 | 0.05 | 0.26 | 294.32 |
| Conv9 | 512 | 2304 | 0.92 | 0.05 | 1.00 | 0.05 | 1.00 | 0.05 | 1.35 | 565.75 |
| Conv10 | 512 | 4608 | 0.93 | 0.05 | 1.00 | 0.05 | 1.00 | 0.05 | 0.95 | 565.75 |
| Conv11 | 512 | 4608 | 0.94 | 0.05 | 1.00 | 0.05 | 1.00 | 0.05 | 0.21 | 565.75 |
| Conv12 | 512 | 4608 | 0.94 | 0.05 | 1.00 | 0.05 | 1.00 | 0.05 | 0.21 | 565.75 |
| Conv13 | 512 | 4608 | 0.95 | 0.05 | 1.00 | 0.05 | 1.00 | 0.05 | 0.19 | 565.75 |
| Conv14 | 512 | 4608 | 0.95 | 0.05 | 1.00 | 0.05 | 1.00 | 0.05 | 0.15 | 565.75 |
| Conv15 | 512 | 4608 | 0.95 | 0.05 | 1.00 | 0.05 | 1.00 | 0.05 | 0.11 | 565.75 |
| Conv16 | 512 | 4608 | 0.93 | 0.05 | 1.00 | 0.05 | 1.00 | 0.05 | 0.37 | 565.75 |
| **Passing rate** | - | - | 100.00% | | 100.00% | | 100.00% | | 93.75% | |

Table 10: Cifar100 AlexNet

| Layer | Number | dim | Gaussian | | Mean_Left | | Mean_Right | | Sigma | |
|---|---|---|---|---|---|---|---|---|---|---|
| | | | p-value | c-value | p-value | c-value | p-value | c-value | t-value | c-value |
| Conv1 | 64 | 363 | 0.51 | 0.05 | 0.94 | 0.05 | 0.94 | 0.05 | 54.46 | 83.68 |
| Conv2 | 192 | 1600 | 0.39 | 0.05 | 1.00 | 0.05 | 1.00 | 0.05 | 44.19 | 225.33 |
| Conv3 | 384 | 1728 | 0.66 | 0.05 | 1.00 | 0.05 | 1.00 | 0.05 | 6.62 | 430.69 |
| Conv4 | 256 | 3456 | 0.87 | 0.05 | 1.00 | 0.05 | 1.00 | 0.05 | 1.40 | 294.32 |
| Conv5 | 256 | 2304 | 0.85 | 0.05 | 1.00 | 0.05 | 1.00 | 0.05 | 8.74 | 294.32 |
| **Passing rate** | - | - | 100.00% | | 100.00% | | 100.00% | | 100.00% | |

Table 11: Cifar100 DenseNet-bc-100-12

| Layer | Number | dim | Gaussian | | Mean_Left | | Mean_Right | | Sigma | |
|---|---|---|---|---|---|---|---|---|---|---|
| | | | p-value | c-value | p-value | c-value | p-value | c-value | t-value | c-value |
| Conv1 | 24 | 27 | 0.06 | 0.05 | 0.60 | 0.05 | 0.60 | 0.05 | 294.0 | 36.42 |
| Conv2 | 48 | 24 | 0.64 | 0.05 | 0.63 | 0.05 | 0.63 | 0.05 | 21.13 | 65.17 |
| Conv3 | 12 | 432 | 0.63 | 0.05 | 0.76 | 0.05 | 0.76 | 0.05 | 0.52 | 21.03 |
| Conv4 | 48 | 36 | 0.63 | 0.05 | 0.66 | 0.05 | 0.66 | 0.05 | 4.04 | 65.17 |
| Conv5 | 12 | 432 | 0.83 | 0.05 | 0.76 | 0.05 | 0.76 | 0.05 | 0.16 | 21.03 |
| Conv6 | 48 | 48 | 0.54 | 0.05 | 0.68 | 0.05 | 0.68 | 0.05 | 12.54 | 65.17 |
| Conv7 | 12 | 432 | 0.62 | 0.05 | 0.76 | 0.05 | 0.76 | 0.05 | 0.61 | 21.03 |
| Conv8 | 48 | 60 | 0.64 | 0.05 | 0.70 | 0.05 | 0.70 | 0.05 | 3.88 | 65.17 |
| Conv9 | 12 | 432 | 0.86 | 0.05 | 0.76 | 0.05 | 0.76 | 0.05 | 0.21 | 21.03 |
| Conv10 | 48 | 72 | 0.67 | 0.05 | 0.72 | 0.05 | 0.72 | 0.05 | 2.61 | 65.17 |
| Conv11 | 12 | 432 | 0.71 | 0.05 | 0.76 | 0.05 | 0.76 | 0.05 | 0.12 | 21.03 |
| Conv12 | 48 | 84 | 0.62 | 0.05 | 0.74 | 0.05 | 0.74 | 0.05 | 3.88 | 65.17 |

| | | | | | | | | | | |
|---|---|---|---|---|---|---|---|---|---|---|
| Conv13 | 12 | 432 | 0.85 | 0.05 | 0.76 | 0.05 | 0.76 | 0.05 | 0.59 | 21.03 |
| Conv14 | 48 | 96 | 0.82 | 0.05 | 0.75 | 0.05 | 0.75 | 0.05 | 1.89 | 65.17 |
| Conv15 | 12 | 432 | 0.86 | 0.05 | 0.76 | 0.05 | 0.76 | 0.05 | 0.27 | 21.03 |
| Conv16 | 48 | 108 | 0.70 | 0.05 | 0.76 | 0.05 | 0.76 | 0.05 | 3.75 | 65.17 |
| Conv17 | 12 | 432 | 0.86 | 0.05 | 0.76 | 0.05 | 0.76 | 0.05 | 0.28 | 21.03 |
| Conv18 | 48 | 120 | 0.85 | 0.05 | 0.78 | 0.05 | 0.78 | 0.05 | 2.31 | 65.17 |
| Conv19 | 12 | 432 | 0.82 | 0.05 | 0.76 | 0.05 | 0.76 | 0.05 | 0.33 | 21.03 |
| Conv20 | 48 | 132 | 0.59 | 0.05 | 0.79 | 0.05 | 0.79 | 0.05 | 2.91 | 65.17 |
| Conv21 | 12 | 432 | 0.71 | 0.05 | 0.76 | 0.05 | 0.76 | 0.05 | 0.61 | 21.03 |
| Conv22 | 48 | 144 | 0.66 | 0.05 | 0.80 | 0.05 | 0.80 | 0.05 | 3.15 | 65.17 |
| Conv23 | 12 | 432 | 0.86 | 0.05 | 0.76 | 0.05 | 0.76 | 0.05 | 0.36 | 21.03 |
| Conv24 | 48 | 156 | 0.64 | 0.05 | 0.81 | 0.05 | 0.81 | 0.05 | 1.35 | 65.17 |
| Conv25 | 12 | 432 | 0.85 | 0.05 | 0.76 | 0.05 | 0.76 | 0.05 | 0.24 | 21.03 |
| Conv26 | 48 | 168 | 0.70 | 0.05 | 0.82 | 0.05 | 0.82 | 0.05 | 1.15 | 65.17 |
| Conv27 | 12 | 432 | 0.88 | 0.05 | 0.76 | 0.05 | 0.76 | 0.05 | 0.09 | 21.03 |
| Conv28 | 48 | 180 | 0.60 | 0.05 | 0.82 | 0.05 | 0.82 | 0.05 | 1.84 | 65.17 |
| Conv29 | 12 | 432 | 0.83 | 0.05 | 0.76 | 0.05 | 0.76 | 0.05 | 0.19 | 21.03 |
| Conv30 | 48 | 192 | 0.69 | 0.05 | 0.83 | 0.05 | 0.83 | 0.05 | 2.52 | 65.17 |
| Conv31 | 12 | 432 | 0.87 | 0.05 | 0.76 | 0.05 | 0.76 | 0.05 | 0.11 | 21.03 |
| Conv32 | 48 | 204 | 0.61 | 0.05 | 0.84 | 0.05 | 0.84 | 0.05 | 1.71 | 65.17 |
| Conv33 | 12 | 432 | 0.83 | 0.05 | 0.76 | 0.05 | 0.76 | 0.05 | 0.33 | 21.03 |
| Conv34 | 108 | 216 | 0.29 | 0.05 | 0.94 | 0.05 | 0.94 | 0.05 | 5.21 | 133.26 |
| Conv35 | 48 | 108 | 0.67 | 0.05 | 0.76 | 0.05 | 0.76 | 0.05 | 1.89 | 65.17 |
| Conv36 | 12 | 432 | 0.70 | 0.05 | 0.76 | 0.05 | 0.76 | 0.05 | 0.41 | 21.03 |
| Conv37 | 48 | 120 | 0.67 | 0.05 | 0.78 | 0.05 | 0.78 | 0.05 | 1.54 | 65.17 |
| Conv38 | 12 | 432 | 0.84 | 0.05 | 0.76 | 0.05 | 0.76 | 0.05 | 0.13 | 21.03 |
| Conv39 | 48 | 132 | 0.83 | 0.05 | 0.79 | 0.05 | 0.79 | 0.05 | 2.06 | 65.17 |
| Conv40 | 12 | 432 | 0.82 | 0.05 | 0.76 | 0.05 | 0.76 | 0.05 | 0.67 | 21.03 |
| Conv41 | 48 | 144 | 0.68 | 0.05 | 0.80 | 0.05 | 0.80 | 0.05 | 1.63 | 65.17 |
| Conv42 | 12 | 432 | 0.81 | 0.05 | 0.76 | 0.05 | 0.76 | 0.05 | 0.29 | 21.03 |
| Conv43 | 48 | 156 | 0.67 | 0.05 | 0.81 | 0.05 | 0.81 | 0.05 | 1.16 | 65.17 |
| Conv44 | 12 | 432 | 0.85 | 0.05 | 0.76 | 0.05 | 0.76 | 0.05 | 0.13 | 21.03 |
| Conv45 | 48 | 168 | 0.65 | 0.05 | 0.82 | 0.05 | 0.82 | 0.05 | 1.71 | 65.17 |
| Conv46 | 12 | 432 | 0.69 | 0.05 | 0.76 | 0.05 | 0.76 | 0.05 | 0.35 | 21.03 |
| Conv47 | 48 | 180 | 0.82 | 0.05 | 0.82 | 0.05 | 0.82 | 0.05 | 1.19 | 65.17 |
| Conv48 | 12 | 432 | 0.83 | 0.05 | 0.76 | 0.05 | 0.76 | 0.05 | 0.14 | 21.03 |
| Conv49 | 48 | 192 | 0.84 | 0.05 | 0.83 | 0.05 | 0.83 | 0.05 | 0.93 | 65.17 |
| Conv50 | 12 | 432 | 0.67 | 0.05 | 0.76 | 0.05 | 0.76 | 0.05 | 0.36 | 21.03 |
| Conv51 | 48 | 204 | 0.81 | 0.05 | 0.84 | 0.05 | 0.84 | 0.05 | 1.11 | 65.17 |
| Conv52 | 12 | 432 | 0.81 | 0.05 | 0.76 | 0.05 | 0.76 | 0.05 | 0.25 | 21.03 |
| Conv53 | 48 | 216 | 0.84 | 0.05 | 0.85 | 0.05 | 0.85 | 0.05 | 0.66 | 65.17 |
| Conv54 | 12 | 432 | 0.83 | 0.05 | 0.76 | 0.05 | 0.76 | 0.05 | 0.2 | 21.03 |
| Conv55 | 48 | 228 | 0.80 | 0.05 | 0.85 | 0.05 | 0.85 | 0.05 | 1.14 | 65.17 |
| Conv56 | 12 | 432 | 0.65 | 0.05 | 0.76 | 0.05 | 0.76 | 0.05 | 0.37 | 21.03 |
| Conv57 | 48 | 240 | 0.82 | 0.05 | 0.86 | 0.05 | 0.86 | 0.05 | 0.91 | 65.17 |
| Conv58 | 12 | 432 | 0.83 | 0.05 | 0.76 | 0.05 | 0.76 | 0.05 | 0.25 | 21.03 |
| Conv59 | 48 | 252 | 0.67 | 0.05 | 0.86 | 0.05 | 0.86 | 0.05 | 0.98 | 65.17 |
| Conv60 | 12 | 432 | 0.66 | 0.05 | 0.76 | 0.05 | 0.76 | 0.05 | 0.41 | 21.03 |
| Conv61 | 48 | 264 | 0.67 | 0.05 | 0.87 | 0.05 | 0.87 | 0.05 | 0.97 | 65.17 |
| Conv62 | 12 | 432 | 0.83 | 0.05 | 0.76 | 0.05 | 0.76 | 0.05 | 0.24 | 21.03 |
| Conv63 | 48 | 276 | 0.83 | 0.05 | 0.88 | 0.05 | 0.88 | 0.05 | 0.66 | 65.17 |
| Conv64 | 12 | 432 | 0.85 | 0.05 | 0.76 | 0.05 | 0.76 | 0.05 | 0.13 | 21.03 |
| Conv65 | 48 | 288 | 0.83 | 0.05 | 0.88 | 0.05 | 0.88 | 0.05 | 0.73 | 65.17 |
| Conv66 | 12 | 432 | 0.67 | 0.05 | 0.76 | 0.05 | 0.76 | 0.05 | 0.15 | 21.03 |
| Conv67 | 150 | 300 | 0.51 | 0.05 | 0.98 | 0.05 | 0.98 | 0.05 | 1.5 | 179.58 |
| Conv68 | 48 | 150 | 0.66 | 0.05 | 0.80 | 0.05 | 0.80 | 0.05 | 0.93 | 65.17 |
| Conv69 | 12 | 432 | 0.69 | 0.05 | 0.76 | 0.05 | 0.76 | 0.05 | 0.34 | 21.03 |
| Conv70 | 48 | 162 | 0.63 | 0.05 | 0.81 | 0.05 | 0.81 | 0.05 | 1.04 | 65.17 |
| Conv71 | 12 | 432 | 0.65 | 0.05 | 0.76 | 0.05 | 0.76 | 0.05 | 0.18 | 21.03 |

| Layer | | | | | | | | | | |
|---|---|---|---|---|---|---|---|---|---|---|
| Conv72 | 48 | 174 | 0.84 | 0.05 | 0.82 | 0.05 | 0.82 | 0.05 | 0.71 | 65.17 |
| Conv73 | 12 | 432 | 0.82 | 0.05 | 0.76 | 0.05 | 0.76 | 0.05 | 0.53 | 21.03 |
| Conv74 | 48 | 186 | 0.80 | 0.05 | 0.83 | 0.05 | 0.83 | 0.05 | 0.74 | 65.17 |
| Conv75 | 12 | 432 | 0.68 | 0.05 | 0.76 | 0.05 | 0.76 | 0.05 | 0.65 | 21.03 |
| Conv76 | 48 | 198 | 0.33 | 0.05 | 0.84 | 0.05 | 0.84 | 0.05 | 0.9 | 65.17 |
| Conv77 | 12 | 432 | 0.63 | 0.05 | 0.76 | 0.05 | 0.76 | 0.05 | 1.07 | 21.03 |
| Conv78 | 48 | 210 | 0.82 | 0.05 | 0.84 | 0.05 | 0.84 | 0.05 | 0.77 | 65.17 |
| Conv79 | 12 | 432 | 0.81 | 0.05 | 0.76 | 0.05 | 0.76 | 0.05 | 0.31 | 21.03 |
| Conv80 | 48 | 222 | 0.71 | 0.05 | 0.85 | 0.05 | 0.85 | 0.05 | 0.48 | 65.17 |
| Conv81 | 12 | 432 | 0.83 | 0.05 | 0.76 | 0.05 | 0.76 | 0.05 | 0.23 | 21.03 |
| Conv82 | 48 | 234 | 0.82 | 0.05 | 0.86 | 0.05 | 0.86 | 0.05 | 0.79 | 65.17 |
| Conv83 | 12 | 432 | 0.84 | 0.05 | 0.76 | 0.05 | 0.76 | 0.05 | 0.37 | 21.03 |
| Conv84 | 48 | 246 | 0.82 | 0.05 | 0.86 | 0.05 | 0.86 | 0.05 | 1.21 | 65.17 |
| Conv85 | 12 | 432 | 0.84 | 0.05 | 0.76 | 0.05 | 0.76 | 0.05 | 0.07 | 21.03 |
| Conv86 | 48 | 258 | 0.84 | 0.05 | 0.87 | 0.05 | 0.87 | 0.05 | 0.85 | 65.17 |
| Conv87 | 12 | 432 | 0.84 | 0.05 | 0.76 | 0.05 | 0.76 | 0.05 | 0.2 | 21.03 |
| Conv88 | 48 | 270 | 0.80 | 0.05 | 0.87 | 0.05 | 0.87 | 0.05 | 0.88 | 65.17 |
| Conv89 | 12 | 432 | 0.84 | 0.05 | 0.76 | 0.05 | 0.76 | 0.05 | 0.05 | 21.03 |
| Conv90 | 48 | 282 | 0.82 | 0.05 | 0.88 | 0.05 | 0.88 | 0.05 | 0.88 | 65.17 |
| Conv91 | 12 | 432 | 0.86 | 0.05 | 0.76 | 0.05 | 0.76 | 0.05 | 0.04 | 21.03 |
| Conv92 | 48 | 294 | 0.83 | 0.05 | 0.88 | 0.05 | 0.88 | 0.05 | 0.75 | 65.17 |
| Conv93 | 12 | 432 | 0.86 | 0.05 | 0.76 | 0.05 | 0.76 | 0.05 | 0.09 | 21.03 |
| Conv94 | 48 | 306 | 0.81 | 0.05 | 0.89 | 0.05 | 0.89 | 0.05 | 0.79 | 65.17 |
| Conv95 | 12 | 432 | 0.85 | 0.05 | 0.76 | 0.05 | 0.76 | 0.05 | 0.04 | 21.03 |
| Conv96 | 48 | 318 | 0.64 | 0.05 | 0.89 | 0.05 | 0.89 | 0.05 | 0.87 | 65.17 |
| Conv97 | 12 | 432 | 0.83 | 0.05 | 0.76 | 0.05 | 0.76 | 0.05 | 0.09 | 21.03 |
| Conv98 | 48 | 330 | 0.82 | 0.05 | 0.90 | 0.05 | 0.90 | 0.05 | 0.94 | 65.17 |
| Conv99 | 12 | 432 | 0.81 | 0.05 | 0.76 | 0.05 | 0.76 | 0.05 | 0.06 | 21.03 |
| **Passing rate** | - | - | 100.0% | | 100.0% | | 100.0% | | 98.99% | |

Table 12: Cifar100 PreResNet110

| Layer | Number | dim | Gaussian | | Mean_Left | | Mean_Right | | Sigma | |
|---|---|---|---|---|---|---|---|---|---|---|
| | | | p-value | c-value | p-value | c-value | p-value | c-value | t-value | c-value |
| Conv1 | 16 | 27 | 0.05 | 0.05 | 0.58 | 0.05 | 0.58 | 0.05 | 96.78 | 26.3 |
| Conv2 | 16 | 144 | 0.47 | 0.05 | 0.68 | 0.05 | 0.68 | 0.05 | 4.7 | 26.3 |
| Conv3 | 16 | 144 | 0.64 | 0.05 | 0.68 | 0.05 | 0.68 | 0.05 | 3.3 | 26.3 |
| Conv4 | 16 | 144 | 0.49 | 0.05 | 0.68 | 0.05 | 0.68 | 0.05 | 2.8 | 26.3 |
| Conv5 | 16 | 144 | 0.58 | 0.05 | 0.68 | 0.05 | 0.68 | 0.05 | 4.53 | 26.3 |
| Conv6 | 16 | 144 | 0.53 | 0.05 | 0.68 | 0.05 | 0.68 | 0.05 | 1.88 | 26.3 |
| Conv7 | 16 | 144 | 0.59 | 0.05 | 0.68 | 0.05 | 0.68 | 0.05 | 1.29 | 26.3 |
| Conv8 | 16 | 144 | 0.67 | 0.05 | 0.68 | 0.05 | 0.68 | 0.05 | 0.69 | 26.3 |
| Conv9 | 16 | 144 | 0.70 | 0.05 | 0.68 | 0.05 | 0.68 | 0.05 | 1.2 | 26.3 |
| Conv10 | 16 | 144 | 0.58 | 0.05 | 0.68 | 0.05 | 0.68 | 0.05 | 2.41 | 26.3 |
| Conv11 | 16 | 144 | 0.62 | 0.05 | 0.68 | 0.05 | 0.68 | 0.05 | 1.5 | 26.3 |
| Conv12 | 16 | 144 | 0.69 | 0.05 | 0.68 | 0.05 | 0.68 | 0.05 | 0.77 | 26.3 |
| Conv13 | 16 | 144 | 0.62 | 0.05 | 0.68 | 0.05 | 0.68 | 0.05 | 2.18 | 26.3 |
| Conv14 | 16 | 144 | 0.69 | 0.05 | 0.68 | 0.05 | 0.68 | 0.05 | 0.63 | 26.3 |
| Conv15 | 16 | 144 | 0.67 | 0.05 | 0.68 | 0.05 | 0.68 | 0.05 | 1.66 | 26.3 |
| Conv16 | 16 | 144 | 0.81 | 0.05 | 0.68 | 0.05 | 0.68 | 0.05 | 0.87 | 26.3 |
| Conv17 | 16 | 144 | 0.67 | 0.05 | 0.68 | 0.05 | 0.68 | 0.05 | 1.22 | 26.3 |
| Conv18 | 16 | 144 | 0.69 | 0.05 | 0.68 | 0.05 | 0.68 | 0.05 | 0.72 | 26.3 |
| Conv19 | 16 | 144 | 0.82 | 0.05 | 0.68 | 0.05 | 0.68 | 0.05 | 1.29 | 26.3 |
| Conv20 | 16 | 144 | 0.66 | 0.05 | 0.68 | 0.05 | 0.68 | 0.05 | 1.07 | 26.3 |
| Conv21 | 16 | 144 | 0.83 | 0.05 | 0.68 | 0.05 | 0.68 | 0.05 | 1.64 | 26.3 |

| | | | | | | | | | | |
|---|---|---|---|---|---|---|---|---|---|---|
| Conv22 | 16 | 144 | 0.69 | 0.05 | 0.68 | 0.05 | 0.68 | 0.05 | 0.93 | 26.3 |
| Conv23 | 16 | 144 | 0.70 | 0.05 | 0.68 | 0.05 | 0.68 | 0.05 | 1.76 | 26.3 |
| Conv24 | 16 | 144 | 0.82 | 0.05 | 0.68 | 0.05 | 0.68 | 0.05 | 0.9 | 26.3 |
| Conv25 | 16 | 144 | 0.83 | 0.05 | 0.68 | 0.05 | 0.68 | 0.05 | 1.51 | 26.3 |
| Conv26 | 16 | 144 | 0.68 | 0.05 | 0.68 | 0.05 | 0.68 | 0.05 | 1.28 | 26.3 |
| Conv27 | 16 | 144 | 0.81 | 0.05 | 0.68 | 0.05 | 0.68 | 0.05 | 2.35 | 26.3 |
| Conv28 | 16 | 144 | 0.68 | 0.05 | 0.68 | 0.05 | 0.68 | 0.05 | 1.25 | 26.3 |
| Conv29 | 16 | 144 | 0.80 | 0.05 | 0.68 | 0.05 | 0.68 | 0.05 | 2.06 | 26.3 |
| Conv30 | 16 | 144 | 0.69 | 0.05 | 0.68 | 0.05 | 0.68 | 0.05 | 2.32 | 26.3 |
| Conv31 | 16 | 144 | 0.85 | 0.05 | 0.68 | 0.05 | 0.68 | 0.05 | 0.81 | 26.3 |
| Conv32 | 16 | 144 | 0.56 | 0.05 | 0.68 | 0.05 | 0.68 | 0.05 | 5.68 | 26.3 |
| Conv33 | 16 | 144 | 0.49 | 0.05 | 0.68 | 0.05 | 0.68 | 0.05 | 9.37 | 26.3 |
| Conv34 | 16 | 144 | 0.58 | 0.05 | 0.68 | 0.05 | 0.68 | 0.05 | 3.73 | 26.3 |
| Conv35 | 16 | 144 | 0.83 | 0.05 | 0.68 | 0.05 | 0.68 | 0.05 | 0.76 | 26.3 |
| Conv36 | 16 | 144 | 0.49 | 0.05 | 0.68 | 0.05 | 0.68 | 0.05 | 7.17 | 26.3 |
| Conv37 | 16 | 144 | 0.55 | 0.05 | 0.68 | 0.05 | 0.68 | 0.05 | 4.83 | 26.3 |
| Conv38 | 32 | 144 | 0.45 | 0.05 | 0.75 | 0.05 | 0.75 | 0.05 | 2.35 | 46.19 |
| Conv39 | 32 | 288 | 0.54 | 0.05 | 0.83 | 0.05 | 0.83 | 0.05 | 1.89 | 46.19 |
| Conv40 | 32 | 288 | 0.63 | 0.05 | 0.83 | 0.05 | 0.83 | 0.05 | 1.25 | 46.19 |
| Conv41 | 32 | 288 | 0.69 | 0.05 | 0.83 | 0.05 | 0.83 | 0.05 | 0.91 | 46.19 |
| Conv42 | 32 | 288 | 0.71 | 0.05 | 0.83 | 0.05 | 0.83 | 0.05 | 0.58 | 46.19 |
| Conv43 | 32 | 288 | 0.83 | 0.05 | 0.83 | 0.05 | 0.83 | 0.05 | 0.45 | 46.19 |
| Conv44 | 32 | 288 | 0.56 | 0.05 | 0.83 | 0.05 | 0.83 | 0.05 | 1.58 | 46.19 |
| Conv45 | 32 | 288 | 0.67 | 0.05 | 0.83 | 0.05 | 0.83 | 0.05 | 0.74 | 46.19 |
| Conv46 | 32 | 288 | 0.81 | 0.05 | 0.83 | 0.05 | 0.83 | 0.05 | 0.81 | 46.19 |
| Conv47 | 32 | 288 | 0.70 | 0.05 | 0.83 | 0.05 | 0.83 | 0.05 | 0.59 | 46.19 |
| Conv48 | 32 | 288 | 0.65 | 0.05 | 0.83 | 0.05 | 0.83 | 0.05 | 1.22 | 46.19 |
| Conv49 | 32 | 288 | 0.80 | 0.05 | 0.83 | 0.05 | 0.83 | 0.05 | 0.34 | 46.19 |
| Conv50 | 32 | 288 | 0.69 | 0.05 | 0.83 | 0.05 | 0.83 | 0.05 | 1.18 | 46.19 |
| Conv51 | 32 | 288 | 0.83 | 0.05 | 0.83 | 0.05 | 0.83 | 0.05 | 0.74 | 46.19 |
| Conv52 | 32 | 288 | 0.67 | 0.05 | 0.83 | 0.05 | 0.83 | 0.05 | 1.18 | 46.19 |
| Conv53 | 32 | 288 | 0.85 | 0.05 | 0.83 | 0.05 | 0.83 | 0.05 | 0.36 | 46.19 |
| Conv54 | 32 | 288 | 0.55 | 0.05 | 0.83 | 0.05 | 0.83 | 0.05 | 2.03 | 46.19 |
| Conv55 | 32 | 288 | 0.82 | 0.05 | 0.83 | 0.05 | 0.83 | 0.05 | 0.88 | 46.19 |
| Conv56 | 32 | 288 | 0.70 | 0.05 | 0.83 | 0.05 | 0.83 | 0.05 | 1.14 | 46.19 |
| Conv57 | 32 | 288 | 0.83 | 0.05 | 0.83 | 0.05 | 0.83 | 0.05 | 0.42 | 46.19 |
| Conv58 | 32 | 288 | 0.69 | 0.05 | 0.83 | 0.05 | 0.83 | 0.05 | 1.51 | 46.19 |
| Conv59 | 32 | 288 | 0.82 | 0.05 | 0.83 | 0.05 | 0.83 | 0.05 | 0.57 | 46.19 |
| Conv60 | 32 | 288 | 0.69 | 0.05 | 0.83 | 0.05 | 0.83 | 0.05 | 0.9 | 46.19 |
| Conv61 | 32 | 288 | 0.85 | 0.05 | 0.83 | 0.05 | 0.83 | 0.05 | 0.43 | 46.19 |
| Conv62 | 32 | 288 | 0.68 | 0.05 | 0.83 | 0.05 | 0.83 | 0.05 | 1.17 | 46.19 |
| Conv63 | 32 | 288 | 0.82 | 0.05 | 0.83 | 0.05 | 0.83 | 0.05 | 0.61 | 46.19 |
| Conv64 | 32 | 288 | 0.60 | 0.05 | 0.83 | 0.05 | 0.83 | 0.05 | 1.18 | 46.19 |
| Conv65 | 32 | 288 | 0.85 | 0.05 | 0.83 | 0.05 | 0.83 | 0.05 | 0.6 | 46.19 |
| Conv66 | 32 | 288 | 0.67 | 0.05 | 0.83 | 0.05 | 0.83 | 0.05 | 1.04 | 46.19 |
| Conv67 | 32 | 288 | 0.85 | 0.05 | 0.83 | 0.05 | 0.83 | 0.05 | 0.41 | 46.19 |
| Conv68 | 32 | 288 | 0.64 | 0.05 | 0.83 | 0.05 | 0.83 | 0.05 | 1.14 | 46.19 |
| Conv69 | 32 | 288 | 0.85 | 0.05 | 0.83 | 0.05 | 0.83 | 0.05 | 0.58 | 46.19 |
| Conv70 | 32 | 288 | 0.69 | 0.05 | 0.83 | 0.05 | 0.83 | 0.05 | 0.9 | 46.19 |
| Conv71 | 32 | 288 | 0.85 | 0.05 | 0.83 | 0.05 | 0.83 | 0.05 | 0.48 | 46.19 |
| Conv72 | 32 | 288 | 0.69 | 0.05 | 0.83 | 0.05 | 0.83 | 0.05 | 1.2 | 46.19 |
| Conv73 | 32 | 288 | 0.84 | 0.05 | 0.83 | 0.05 | 0.83 | 0.05 | 0.38 | 46.19 |
| Conv74 | 64 | 288 | 0.41 | 0.05 | 0.91 | 0.05 | 0.91 | 0.05 | 1.27 | 83.68 |
| Conv75 | 64 | 576 | 0.50 | 0.05 | 0.97 | 0.05 | 0.97 | 0.05 | 1.01 | 83.68 |
| Conv76 | 64 | 576 | 0.68 | 0.05 | 0.97 | 0.05 | 0.97 | 0.05 | 1.39 | 83.68 |
| Conv77 | 64 | 576 | 0.80 | 0.05 | 0.97 | 0.05 | 0.97 | 0.05 | 0.96 | 83.68 |
| Conv78 | 64 | 576 | 0.64 | 0.05 | 0.97 | 0.05 | 0.97 | 0.05 | 1.26 | 83.68 |
| Conv79 | 64 | 576 | 0.84 | 0.05 | 0.97 | 0.05 | 0.97 | 0.05 | 0.99 | 83.68 |
| Conv80 | 64 | 576 | 0.71 | 0.05 | 0.97 | 0.05 | 0.97 | 0.05 | 0.83 | 83.68 |

| Layer | Number | dim | | | | | | | | |
|-------|--------|-----|------|------|------|------|------|------|------|------|
| Conv81 | 64 | 576 | 0.82 | 0.05 | 0.97 | 0.05 | 0.97 | 0.05 | 0.74 | 83.68 |
| Conv82 | 64 | 576 | 0.64 | 0.05 | 0.97 | 0.05 | 0.97 | 0.05 | 1.46 | 83.68 |
| Conv83 | 64 | 576 | 0.71 | 0.05 | 0.97 | 0.05 | 0.97 | 0.05 | 0.75 | 83.68 |
| Conv84 | 64 | 576 | 0.81 | 0.05 | 0.97 | 0.05 | 0.97 | 0.05 | 1.02 | 83.68 |
| Conv85 | 64 | 576 | 0.84 | 0.05 | 0.97 | 0.05 | 0.97 | 0.05 | 0.64 | 83.68 |
| Conv86 | 64 | 576 | 0.70 | 0.05 | 0.97 | 0.05 | 0.97 | 0.05 | 1.08 | 83.68 |
| Conv87 | 64 | 576 | 0.85 | 0.05 | 0.97 | 0.05 | 0.97 | 0.05 | 0.44 | 83.68 |
| Conv88 | 64 | 576 | 0.81 | 0.05 | 0.97 | 0.05 | 0.97 | 0.05 | 0.71 | 83.68 |
| Conv89 | 64 | 576 | 0.87 | 0.05 | 0.97 | 0.05 | 0.97 | 0.05 | 0.53 | 83.68 |
| Conv90 | 64 | 576 | 0.65 | 0.05 | 0.97 | 0.05 | 0.97 | 0.05 | 0.89 | 83.68 |
| Conv91 | 64 | 576 | 0.85 | 0.05 | 0.97 | 0.05 | 0.97 | 0.05 | 0.39 | 83.68 |
| Conv92 | 64 | 576 | 0.80 | 0.05 | 0.97 | 0.05 | 0.97 | 0.05 | 0.83 | 83.68 |
| Conv93 | 64 | 576 | 0.85 | 0.05 | 0.97 | 0.05 | 0.97 | 0.05 | 0.4 | 83.68 |
| Conv94 | 64 | 576 | 0.83 | 0.05 | 0.97 | 0.05 | 0.97 | 0.05 | 0.58 | 83.68 |
| Conv95 | 64 | 576 | 0.88 | 0.05 | 0.97 | 0.05 | 0.97 | 0.05 | 0.38 | 83.68 |
| Conv96 | 64 | 576 | 0.86 | 0.05 | 0.97 | 0.05 | 0.97 | 0.05 | 0.77 | 83.68 |
| Conv97 | 64 | 576 | 0.88 | 0.05 | 0.97 | 0.05 | 0.97 | 0.05 | 0.37 | 83.68 |
| Conv98 | 64 | 576 | 0.86 | 0.05 | 0.97 | 0.05 | 0.97 | 0.05 | 0.76 | 83.68 |
| Conv99 | 64 | 576 | 0.87 | 0.05 | 0.97 | 0.05 | 0.97 | 0.05 | 0.35 | 83.68 |
| Conv100 | 64 | 576 | 0.87 | 0.05 | 0.97 | 0.05 | 0.97 | 0.05 | 0.79 | 83.68 |
| Conv101 | 64 | 576 | 0.87 | 0.05 | 0.97 | 0.05 | 0.97 | 0.05 | 0.42 | 83.68 |
| Conv102 | 64 | 576 | 0.87 | 0.05 | 0.97 | 0.05 | 0.97 | 0.05 | 0.62 | 83.68 |
| Conv103 | 64 | 576 | 0.88 | 0.05 | 0.97 | 0.05 | 0.97 | 0.05 | 0.42 | 83.68 |
| Conv104 | 64 | 576 | 0.84 | 0.05 | 0.97 | 0.05 | 0.97 | 0.05 | 0.6 | 83.68 |
| Conv105 | 64 | 576 | 0.89 | 0.05 | 0.97 | 0.05 | 0.97 | 0.05 | 0.55 | 83.68 |
| Conv106 | 64 | 576 | 0.84 | 0.05 | 0.97 | 0.05 | 0.97 | 0.05 | 0.62 | 83.68 |
| Conv107 | 64 | 576 | 0.87 | 0.05 | 0.97 | 0.05 | 0.97 | 0.05 | 0.6 | 83.68 |
| Conv108 | 64 | 576 | 0.87 | 0.05 | 0.97 | 0.05 | 0.97 | 0.05 | 0.38 | 83.68 |
| Conv109 | 64 | 576 | 0.88 | 0.05 | 0.97 | 0.05 | 0.97 | 0.05 | 0.47 | 83.68 |
| **Passing rate** | - | - | 100.0% | | 100.0% | | 100.0% | | 99.08% | |

Table 13: Cifar100 WRN28-10

| Layer | Number | dim | Gaussian | | Mean_Left | | Mean_Right | | Sigma | |
|-------|--------|-----|---------|---------|-----------|---------|------------|---------|-------|---------|
| | | | p-value | c-value | p-value | c-value | p-value | c-value | t-value | c-value |
| Conv1 | 16 | 27 | 0.44 | 0.05 | 0.58 | 0.05 | 0.58 | 0.05 | 24.87 | 26.3 |
| Conv2 | 160 | 144 | 0.92 | 0.05 | 0.94 | 0.05 | 0.94 | 0.05 | 5.96 | 190.52 |
| Conv3 | 160 | 1440 | 0.94 | 0.05 | 1.00 | 0.05 | 1.00 | 0.05 | 0.1 | 190.52 |
| Conv4 | 160 | 16 | 0.67 | 0.05 | 0.69 | 0.05 | 0.69 | 0.05 | 35.0 | 190.52 |
| Conv5 | 160 | 1440 | 0.94 | 0.05 | 1.00 | 0.05 | 1.00 | 0.05 | 0.48 | 190.52 |
| Conv6 | 160 | 1440 | 0.94 | 0.05 | 1.00 | 0.05 | 1.00 | 0.05 | 0.12 | 190.52 |
| Conv7 | 160 | 1440 | 0.94 | 0.05 | 1.00 | 0.05 | 1.00 | 0.05 | 0.03 | 190.52 |
| Conv8 | 160 | 1440 | 0.95 | 0.05 | 1.00 | 0.05 | 1.00 | 0.05 | 0.23 | 190.52 |
| Conv9 | 160 | 1440 | 0.95 | 0.05 | 1.00 | 0.05 | 1.00 | 0.05 | 0.02 | 190.52 |
| Conv10 | 160 | 1440 | 0.95 | 0.05 | 1.00 | 0.05 | 1.00 | 0.05 | 0.39 | 190.52 |
| Conv11 | 320 | 1440 | 0.95 | 0.05 | 1.00 | 0.05 | 1.00 | 0.05 | 1.13 | 362.72 |
| Conv12 | 320 | 2880 | 0.95 | 0.05 | 1.00 | 0.05 | 1.00 | 0.05 | 0.08 | 362.72 |
| Conv13 | 320 | 160 | 0.88 | 0.05 | 0.99 | 0.05 | 0.99 | 0.05 | 2.81 | 362.72 |
| Conv14 | 320 | 2880 | 0.94 | 0.05 | 1.00 | 0.05 | 1.00 | 0.05 | 0.06 | 362.72 |
| Conv15 | 320 | 2880 | 0.95 | 0.05 | 1.00 | 0.05 | 1.00 | 0.05 | 0.06 | 362.72 |
| Conv16 | 320 | 2880 | 0.95 | 0.05 | 1.00 | 0.05 | 1.00 | 0.05 | 0.03 | 362.72 |
| Conv17 | 320 | 2880 | 0.95 | 0.05 | 1.00 | 0.05 | 1.00 | 0.05 | 0.12 | 362.72 |
| Conv18 | 320 | 2880 | 0.95 | 0.05 | 1.00 | 0.05 | 1.00 | 0.05 | 0.04 | 362.72 |
| Conv19 | 320 | 2880 | 0.95 | 0.05 | 1.00 | 0.05 | 1.00 | 0.05 | 0.3 | 362.72 |
| Conv20 | 640 | 2880 | 0.94 | 0.05 | 1.00 | 0.05 | 1.00 | 0.05 | 2.18 | 699.96 |

| Layer | Number | dim | p-value | c-value | p-value | c-value | p-value | c-value | t-value | c-value |
|---|---|---|---|---|---|---|---|---|---|---|
| Conv21 | 640 | 5760 | 0.95 | 0.05 | 1.00 | 0.05 | 1.00 | 0.05 | 0.26 | 699.96 |
| Conv22 | 640 | 320 | 0.91 | 0.05 | 1.00 | 0.05 | 1.00 | 0.05 | 1.91 | 699.96 |
| Conv23 | 640 | 5760 | 0.94 | 0.05 | 1.00 | 0.05 | 1.00 | 0.05 | 0.35 | 699.96 |
| Conv24 | 640 | 5760 | 0.94 | 0.05 | 1.00 | 0.05 | 1.00 | 0.05 | 0.24 | 699.96 |
| Conv25 | 640 | 5760 | 0.94 | 0.05 | 1.00 | 0.05 | 1.00 | 0.05 | 0.06 | 699.96 |
| Conv26 | 640 | 5760 | 0.95 | 0.05 | 1.00 | 0.05 | 1.00 | 0.05 | 0.36 | 699.96 |
| Conv27 | 640 | 5760 | 0.95 | 0.05 | 1.00 | 0.05 | 1.00 | 0.05 | 0.04 | 699.96 |
| Conv28 | 640 | 5760 | 0.94 | 0.05 | 1.00 | 0.05 | 1.00 | 0.05 | 1.23 | 699.96 |
| **Passing rate** | - | - | 100.0% | | 100.0% | | 100.0% | | 100.0% | |

Table 14: Cifar100 ResNext-16x64d

| Layer | Number | dim | Gaussian | | Mean_Left | | Mean_Right | | Sigma | |
|---|---|---|---|---|---|---|---|---|---|---|
| | | | p-value | c-value | p-value | c-value | p-value | c-value | t-value | c-value |
| Conv1 | 64 | 27 | 0.70 | 0.05 | 0.66 | 0.05 | 0.66 | 0.05 | 32.22 | 83.68 |
| Conv2 | 1024 | 64 | 0.91 | 0.05 | 0.99 | 0.05 | 0.99 | 0.05 | 1.51 | 1099.56 |
| Conv3 | 1024 | 576 | 0.91 | 0.05 | 1.00 | 0.05 | 1.00 | 0.05 | 0.41 | 1099.56 |
| Conv4 | 256 | 1024 | 0.93 | 0.05 | 1.00 | 0.05 | 1.00 | 0.05 | 0.12 | 294.32 |
| Conv5 | 256 | 64 | 0.86 | 0.05 | 0.90 | 0.05 | 0.90 | 0.05 | 5.91 | 294.32 |
| Conv6 | 1024 | 256 | 0.95 | 0.05 | 1.00 | 0.05 | 1.00 | 0.05 | 0.13 | 1099.56 |
| Conv7 | 1024 | 576 | 0.95 | 0.05 | 1.00 | 0.05 | 1.00 | 0.05 | 0.04 | 1099.56 |
| Conv8 | 256 | 1024 | 0.94 | 0.05 | 1.00 | 0.05 | 1.00 | 0.05 | 0.03 | 294.32 |
| Conv9 | 1024 | 256 | 0.93 | 0.05 | 1.00 | 0.05 | 1.00 | 0.05 | 0.77 | 1099.56 |
| Conv10 | 1024 | 576 | 0.93 | 0.05 | 1.00 | 0.05 | 1.00 | 0.05 | 0.46 | 1099.56 |
| Conv11 | 256 | 1024 | 0.95 | 0.05 | 1.00 | 0.05 | 1.00 | 0.05 | 0.22 | 294.32 |
| Conv12 | 2048 | 256 | 0.95 | 0.05 | 1.00 | 0.05 | 1.00 | 0.05 | 1.79 | 2154.4 |
| Conv13 | 2048 | 1152 | 0.96 | 0.05 | 1.00 | 0.05 | 1.00 | 0.05 | 0.55 | 2154.4 |
| Conv14 | 512 | 2048 | 0.94 | 0.05 | 1.00 | 0.05 | 1.00 | 0.05 | 0.6 | 565.75 |
| Conv15 | 512 | 256 | 0.93 | 0.05 | 1.00 | 0.05 | 1.00 | 0.05 | 2.92 | 565.75 |
| Conv16 | 2048 | 512 | 0.94 | 0.05 | 1.00 | 0.05 | 1.00 | 0.05 | 0.94 | 2154.4 |
| Conv17 | 2048 | 1152 | 0.95 | 0.05 | 1.00 | 0.05 | 1.00 | 0.05 | 0.82 | 2154.4 |
| Conv18 | 512 | 2048 | 0.94 | 0.05 | 1.00 | 0.05 | 1.00 | 0.05 | 0.4 | 565.75 |
| Conv19 | 2048 | 512 | 0.93 | 0.05 | 1.00 | 0.05 | 1.00 | 0.05 | 1.55 | 2154.4 |
| Conv20 | 2048 | 1152 | 0.95 | 0.05 | 1.00 | 0.05 | 1.00 | 0.05 | 0.95 | 2154.4 |
| Conv21 | 512 | 2048 | 0.95 | 0.05 | 1.00 | 0.05 | 1.00 | 0.05 | 0.39 | 565.75 |
| Conv22 | 4096 | 512 | 0.92 | 0.05 | 1.00 | 0.05 | 1.00 | 0.05 | 3.13 | 4246.0 |
| Conv23 | 4096 | 2304 | 0.95 | 0.05 | 1.00 | 0.05 | 1.00 | 0.05 | 0.65 | 4246.0 |
| Conv24 | 1024 | 4096 | 0.92 | 0.05 | 1.00 | 0.05 | 1.00 | 0.05 | 0.64 | 1099.56 |
| Conv25 | 1024 | 512 | 0.94 | 0.05 | 1.00 | 0.05 | 1.00 | 0.05 | 2.9 | 1099.56 |
| Conv26 | 4096 | 1024 | 0.88 | 0.05 | 1.00 | 0.05 | 1.00 | 0.05 | 1.72 | 4246.0 |
| Conv27 | 4096 | 2304 | 0.94 | 0.05 | 1.00 | 0.05 | 1.00 | 0.05 | 1.54 | 4246.0 |
| Conv28 | 1024 | 4096 | 0.94 | 0.05 | 1.00 | 0.05 | 1.00 | 0.05 | 0.24 | 1099.56 |
| Conv29 | 4096 | 1024 | 0.95 | 0.05 | 1.00 | 0.05 | 1.00 | 0.05 | 2.25 | 4246.0 |
| Conv30 | 4096 | 2304 | 0.96 | 0.05 | 1.00 | 0.05 | 1.00 | 0.05 | 1.03 | 4246.0 |
| Conv31 | 1024 | 4096 | 0.94 | 0.05 | 1.00 | 0.05 | 1.00 | 0.05 | 0.22 | 1099.56 |
| **Passing rate** | - | - | 100.0% | | 100.0% | | 100.0% | | 100.0% | |

P.2   OPTIMIZER

config:

```
https://github.com/bearpaw/pytorch-classification.
```

```
https://pytorch.org/docs/master/optim.html#torch-optim.
```

Table 15: Cifar100 ASGD-ResNet164

| Layer | Number | dim | Gaussian | | Mean_Left | | Mean_Right | | Sigma | |
|-------|--------|-----|---------|---------|---------|---------|---------|---------|---------|---------|
| | | | p-value | c-value | p-value | c-value | p-value | c-value | t-value | c-value |
| Conv1 | 16 | 27 | 0.13 | 0.05 | 0.58 | 0.05 | 0.58 | 0.05 | 34.35 | 26.3 |
| Conv2 | 16 | 16 | 0.33 | 0.05 | 0.56 | 0.05 | 0.56 | 0.05 | 9.29 | 26.3 |
| Conv3 | 16 | 144 | 0.65 | 0.05 | 0.68 | 0.05 | 0.68 | 0.05 | 0.42 | 26.3 |
| Conv4 | 64 | 16 | 0.58 | 0.05 | 0.63 | 0.05 | 0.63 | 0.05 | 13.54 | 83.68 |
| Conv5 | 16 | 64 | 0.24 | 0.05 | 0.63 | 0.05 | 0.63 | 0.05 | 2.95 | 26.3 |
| Conv6 | 16 | 144 | 0.68 | 0.05 | 0.68 | 0.05 | 0.68 | 0.05 | 0.21 | 26.3 |
| Conv7 | 64 | 16 | 0.58 | 0.05 | 0.63 | 0.05 | 0.63 | 0.05 | 16.09 | 83.68 |
| Conv8 | 16 | 64 | 0.29 | 0.05 | 0.63 | 0.05 | 0.63 | 0.05 | 4.05 | 26.3 |
| Conv9 | 16 | 144 | 0.65 | 0.05 | 0.68 | 0.05 | 0.68 | 0.05 | 0.15 | 26.3 |
| Conv10 | 64 | 16 | 0.60 | 0.05 | 0.63 | 0.05 | 0.63 | 0.05 | 15.79 | 83.68 |
| Conv11 | 16 | 64 | 0.29 | 0.05 | 0.63 | 0.05 | 0.63 | 0.05 | 5.69 | 26.3 |
| Conv12 | 16 | 144 | 0.70 | 0.05 | 0.68 | 0.05 | 0.68 | 0.05 | 0.28 | 26.3 |
| Conv13 | 64 | 16 | 0.61 | 0.05 | 0.63 | 0.05 | 0.63 | 0.05 | 20.88 | 83.68 |
| Conv14 | 16 | 64 | 0.29 | 0.05 | 0.63 | 0.05 | 0.63 | 0.05 | 8.67 | 26.3 |
| Conv15 | 16 | 144 | 0.65 | 0.05 | 0.68 | 0.05 | 0.68 | 0.05 | 0.18 | 26.3 |
| Conv16 | 64 | 16 | 0.60 | 0.05 | 0.63 | 0.05 | 0.63 | 0.05 | 17.85 | 83.68 |
| Conv17 | 16 | 64 | 0.29 | 0.05 | 0.63 | 0.05 | 0.63 | 0.05 | 5.21 | 26.3 |
| Conv18 | 16 | 144 | 0.65 | 0.05 | 0.68 | 0.05 | 0.68 | 0.05 | 0.23 | 26.3 |
| Conv19 | 64 | 16 | 0.49 | 0.05 | 0.63 | 0.05 | 0.63 | 0.05 | 21.6 | 83.68 |
| Conv20 | 16 | 64 | 0.24 | 0.05 | 0.63 | 0.05 | 0.63 | 0.05 | 2.93 | 26.3 |
| Conv21 | 16 | 144 | 0.81 | 0.05 | 0.68 | 0.05 | 0.68 | 0.05 | 0.15 | 26.3 |
| Conv22 | 64 | 16 | 0.58 | 0.05 | 0.63 | 0.05 | 0.63 | 0.05 | 15.96 | 83.68 |
| Conv23 | 16 | 64 | 0.30 | 0.05 | 0.63 | 0.05 | 0.63 | 0.05 | 4.21 | 26.3 |
| Conv24 | 16 | 144 | 0.71 | 0.05 | 0.68 | 0.05 | 0.68 | 0.05 | 0.22 | 26.3 |
| Conv25 | 64 | 16 | 0.58 | 0.05 | 0.63 | 0.05 | 0.63 | 0.05 | 15.38 | 83.68 |
| Conv26 | 16 | 64 | 0.29 | 0.05 | 0.63 | 0.05 | 0.63 | 0.05 | 5.46 | 26.3 |
| Conv27 | 16 | 144 | 0.81 | 0.05 | 0.68 | 0.05 | 0.68 | 0.05 | 0.2 | 26.3 |
| Conv28 | 64 | 16 | 0.60 | 0.05 | 0.63 | 0.05 | 0.63 | 0.05 | 17.53 | 83.68 |
| Conv29 | 16 | 64 | 0.36 | 0.05 | 0.63 | 0.05 | 0.63 | 0.05 | 3.99 | 26.3 |
| Conv30 | 16 | 144 | 0.70 | 0.05 | 0.68 | 0.05 | 0.68 | 0.05 | 0.25 | 26.3 |
| Conv31 | 64 | 16 | 0.58 | 0.05 | 0.63 | 0.05 | 0.63 | 0.05 | 17.13 | 83.68 |
| Conv32 | 16 | 64 | 0.24 | 0.05 | 0.63 | 0.05 | 0.63 | 0.05 | 10.61 | 26.3 |
| Conv33 | 16 | 144 | 0.71 | 0.05 | 0.68 | 0.05 | 0.68 | 0.05 | 0.21 | 26.3 |
| Conv34 | 64 | 16 | 0.63 | 0.05 | 0.63 | 0.05 | 0.63 | 0.05 | 21.96 | 83.68 |
| Conv35 | 16 | 64 | 0.18 | 0.05 | 0.63 | 0.05 | 0.63 | 0.05 | 3.58 | 26.3 |
| Conv36 | 16 | 144 | 0.68 | 0.05 | 0.68 | 0.05 | 0.68 | 0.05 | 0.33 | 26.3 |
| Conv37 | 64 | 16 | 0.46 | 0.05 | 0.63 | 0.05 | 0.63 | 0.05 | 21.7 | 83.68 |
| Conv38 | 16 | 64 | 0.28 | 0.05 | 0.63 | 0.05 | 0.63 | 0.05 | 5.06 | 26.3 |
| Conv39 | 16 | 144 | 0.70 | 0.05 | 0.68 | 0.05 | 0.68 | 0.05 | 0.27 | 26.3 |
| Conv40 | 64 | 16 | 0.60 | 0.05 | 0.63 | 0.05 | 0.63 | 0.05 | 23.91 | 83.68 |
| Conv41 | 16 | 64 | 0.30 | 0.05 | 0.63 | 0.05 | 0.63 | 0.05 | 5.54 | 26.3 |
| Conv42 | 16 | 144 | 0.81 | 0.05 | 0.68 | 0.05 | 0.68 | 0.05 | 0.27 | 26.3 |
| Conv43 | 64 | 16 | 0.59 | 0.05 | 0.63 | 0.05 | 0.63 | 0.05 | 20.08 | 83.68 |
| Conv44 | 16 | 64 | 0.24 | 0.05 | 0.63 | 0.05 | 0.63 | 0.05 | 2.2 | 26.3 |
| Conv45 | 16 | 144 | 0.64 | 0.05 | 0.68 | 0.05 | 0.68 | 0.05 | 0.2 | 26.3 |
| Conv46 | 64 | 16 | 0.57 | 0.05 | 0.63 | 0.05 | 0.63 | 0.05 | 15.52 | 83.68 |
| Conv47 | 16 | 64 | 0.31 | 0.05 | 0.63 | 0.05 | 0.63 | 0.05 | 1.59 | 26.3 |
| Conv48 | 16 | 144 | 0.67 | 0.05 | 0.68 | 0.05 | 0.68 | 0.05 | 0.23 | 26.3 |
| Conv49 | 64 | 16 | 0.60 | 0.05 | 0.63 | 0.05 | 0.63 | 0.05 | 22.5 | 83.68 |
| Conv50 | 16 | 64 | 0.27 | 0.05 | 0.63 | 0.05 | 0.63 | 0.05 | 4.37 | 26.3 |
| Conv51 | 16 | 144 | 0.68 | 0.05 | 0.68 | 0.05 | 0.68 | 0.05 | 0.07 | 26.3 |
| Conv52 | 64 | 16 | 0.61 | 0.05 | 0.63 | 0.05 | 0.63 | 0.05 | 20.81 | 83.68 |
| Conv53 | 16 | 64 | 0.27 | 0.05 | 0.63 | 0.05 | 0.63 | 0.05 | 5.5 | 26.3 |

| | | | | | | | | | | |
|---|---|---|---|---|---|---|---|---|---|---|
| Conv54 | 16 | 144 | 0.67 | 0.05 | 0.68 | 0.05 | 0.68 | 0.05 | 0.36 | 26.3 |
| Conv55 | 64 | 16 | 0.57 | 0.05 | 0.63 | 0.05 | 0.63 | 0.05 | 20.01 | 83.68 |
| Conv56 | 32 | 64 | 0.48 | 0.05 | 0.67 | 0.05 | 0.67 | 0.05 | 4.04 | 46.19 |
| Conv57 | 32 | 288 | 0.83 | 0.05 | 0.83 | 0.05 | 0.83 | 0.05 | 0.13 | 46.19 |
| Conv58 | 128 | 32 | 0.66 | 0.05 | 0.74 | 0.05 | 0.74 | 0.05 | 10.05 | 155.4 |
| Conv59 | 32 | 128 | 0.40 | 0.05 | 0.74 | 0.05 | 0.74 | 0.05 | 3.91 | 46.19 |
| Conv60 | 32 | 288 | 0.82 | 0.05 | 0.83 | 0.05 | 0.83 | 0.05 | 0.2 | 46.19 |
| Conv61 | 128 | 32 | 0.68 | 0.05 | 0.74 | 0.05 | 0.74 | 0.05 | 9.64 | 155.4 |
| Conv62 | 32 | 128 | 0.31 | 0.05 | 0.74 | 0.05 | 0.74 | 0.05 | 2.19 | 46.19 |
| Conv63 | 32 | 288 | 0.83 | 0.05 | 0.83 | 0.05 | 0.83 | 0.05 | 0.15 | 46.19 |
| Conv64 | 128 | 32 | 0.66 | 0.05 | 0.74 | 0.05 | 0.74 | 0.05 | 10.3 | 155.4 |
| Conv65 | 32 | 128 | 0.40 | 0.05 | 0.74 | 0.05 | 0.74 | 0.05 | 3.61 | 46.19 |
| Conv66 | 32 | 288 | 0.84 | 0.05 | 0.83 | 0.05 | 0.83 | 0.05 | 0.12 | 46.19 |
| Conv67 | 128 | 32 | 0.60 | 0.05 | 0.74 | 0.05 | 0.74 | 0.05 | 10.61 | 155.4 |
| Conv68 | 32 | 128 | 0.37 | 0.05 | 0.74 | 0.05 | 0.74 | 0.05 | 1.97 | 46.19 |
| Conv69 | 32 | 288 | 0.85 | 0.05 | 0.83 | 0.05 | 0.83 | 0.05 | 0.14 | 46.19 |
| Conv70 | 128 | 32 | 0.70 | 0.05 | 0.74 | 0.05 | 0.74 | 0.05 | 9.04 | 155.4 |
| Conv71 | 32 | 128 | 0.44 | 0.05 | 0.74 | 0.05 | 0.74 | 0.05 | 2.21 | 46.19 |
| Conv72 | 32 | 288 | 0.81 | 0.05 | 0.83 | 0.05 | 0.83 | 0.05 | 0.19 | 46.19 |
| Conv73 | 128 | 32 | 0.68 | 0.05 | 0.74 | 0.05 | 0.74 | 0.05 | 9.69 | 155.4 |
| Conv74 | 32 | 128 | 0.40 | 0.05 | 0.74 | 0.05 | 0.74 | 0.05 | 1.31 | 46.19 |
| Conv75 | 32 | 288 | 0.83 | 0.05 | 0.83 | 0.05 | 0.83 | 0.05 | 0.1 | 46.19 |
| Conv76 | 128 | 32 | 0.68 | 0.05 | 0.74 | 0.05 | 0.74 | 0.05 | 9.75 | 155.4 |
| Conv77 | 32 | 128 | 0.43 | 0.05 | 0.74 | 0.05 | 0.74 | 0.05 | 2.52 | 46.19 |
| Conv78 | 32 | 288 | 0.84 | 0.05 | 0.83 | 0.05 | 0.83 | 0.05 | 0.17 | 46.19 |
| Conv79 | 128 | 32 | 0.67 | 0.05 | 0.74 | 0.05 | 0.74 | 0.05 | 10.85 | 155.4 |
| Conv80 | 32 | 128 | 0.38 | 0.05 | 0.74 | 0.05 | 0.74 | 0.05 | 2.17 | 46.19 |
| Conv81 | 32 | 288 | 0.84 | 0.05 | 0.83 | 0.05 | 0.83 | 0.05 | 0.12 | 46.19 |
| Conv82 | 128 | 32 | 0.68 | 0.05 | 0.74 | 0.05 | 0.74 | 0.05 | 9.48 | 155.4 |
| Conv83 | 32 | 128 | 0.41 | 0.05 | 0.74 | 0.05 | 0.74 | 0.05 | 3.89 | 46.19 |
| Conv84 | 32 | 288 | 0.84 | 0.05 | 0.83 | 0.05 | 0.83 | 0.05 | 0.16 | 46.19 |
| Conv85 | 128 | 32 | 0.62 | 0.05 | 0.74 | 0.05 | 0.74 | 0.05 | 11.02 | 155.4 |
| Conv86 | 32 | 128 | 0.39 | 0.05 | 0.74 | 0.05 | 0.74 | 0.05 | 2.04 | 46.19 |
| Conv87 | 32 | 288 | 0.82 | 0.05 | 0.83 | 0.05 | 0.83 | 0.05 | 0.11 | 46.19 |
| Conv88 | 128 | 32 | 0.66 | 0.05 | 0.74 | 0.05 | 0.74 | 0.05 | 10.19 | 155.4 |
| Conv89 | 32 | 128 | 0.37 | 0.05 | 0.74 | 0.05 | 0.74 | 0.05 | 2.54 | 46.19 |
| Conv90 | 32 | 288 | 0.85 | 0.05 | 0.83 | 0.05 | 0.83 | 0.05 | 0.09 | 46.19 |
| Conv91 | 128 | 32 | 0.65 | 0.05 | 0.74 | 0.05 | 0.74 | 0.05 | 10.0 | 155.4 |
| Conv92 | 32 | 128 | 0.40 | 0.05 | 0.74 | 0.05 | 0.74 | 0.05 | 2.28 | 46.19 |
| Conv93 | 32 | 288 | 0.84 | 0.05 | 0.83 | 0.05 | 0.83 | 0.05 | 0.06 | 46.19 |
| Conv94 | 128 | 32 | 0.69 | 0.05 | 0.74 | 0.05 | 0.74 | 0.05 | 10.16 | 155.4 |
| Conv95 | 32 | 128 | 0.42 | 0.05 | 0.74 | 0.05 | 0.74 | 0.05 | 2.08 | 46.19 |
| Conv96 | 32 | 288 | 0.85 | 0.05 | 0.83 | 0.05 | 0.83 | 0.05 | 0.11 | 46.19 |
| Conv97 | 128 | 32 | 0.64 | 0.05 | 0.74 | 0.05 | 0.74 | 0.05 | 10.08 | 155.4 |
| Conv98 | 32 | 128 | 0.33 | 0.05 | 0.74 | 0.05 | 0.74 | 0.05 | 2.23 | 46.19 |
| Conv99 | 32 | 288 | 0.84 | 0.05 | 0.83 | 0.05 | 0.83 | 0.05 | 0.11 | 46.19 |
| Conv100 | 128 | 32 | 0.66 | 0.05 | 0.74 | 0.05 | 0.74 | 0.05 | 10.11 | 155.4 |
| Conv101 | 32 | 128 | 0.44 | 0.05 | 0.74 | 0.05 | 0.74 | 0.05 | 3.08 | 46.19 |
| Conv102 | 32 | 288 | 0.84 | 0.05 | 0.83 | 0.05 | 0.83 | 0.05 | 0.09 | 46.19 |
| Conv103 | 128 | 32 | 0.65 | 0.05 | 0.74 | 0.05 | 0.74 | 0.05 | 11.58 | 155.4 |
| Conv104 | 32 | 128 | 0.38 | 0.05 | 0.74 | 0.05 | 0.74 | 0.05 | 3.37 | 46.19 |
| Conv105 | 32 | 288 | 0.85 | 0.05 | 0.83 | 0.05 | 0.83 | 0.05 | 0.1 | 46.19 |
| Conv106 | 128 | 32 | 0.68 | 0.05 | 0.74 | 0.05 | 0.74 | 0.05 | 11.59 | 155.4 |
| Conv107 | 32 | 128 | 0.43 | 0.05 | 0.74 | 0.05 | 0.74 | 0.05 | 1.57 | 46.19 |
| Conv108 | 32 | 288 | 0.85 | 0.05 | 0.83 | 0.05 | 0.83 | 0.05 | 0.11 | 46.19 |
| Conv109 | 128 | 32 | 0.66 | 0.05 | 0.74 | 0.05 | 0.74 | 0.05 | 8.07 | 155.4 |
| Conv110 | 64 | 128 | 0.56 | 0.05 | 0.82 | 0.05 | 0.82 | 0.05 | 3.13 | 83.68 |
| Conv111 | 64 | 576 | 0.88 | 0.05 | 0.97 | 0.05 | 0.97 | 0.05 | 0.08 | 83.68 |
| Conv112 | 256 | 64 | 0.84 | 0.05 | 0.90 | 0.05 | 0.90 | 0.05 | 4.87 | 294.32 |

| Conv113 | 64 | 256 | 0.55 | 0.05 | 0.90 | 0.05 | 0.90 | 0.05 | 1.13 | 83.68 |
|---------|-----|-----|------|------|------|------|------|------|------|--------|
| Conv114 | 64 | 576 | 0.87 | 0.05 | 0.97 | 0.05 | 0.97 | 0.05 | 0.07 | 83.68 |
| Conv115 | 256 | 64 | 0.83 | 0.05 | 0.90 | 0.05 | 0.90 | 0.05 | 4.3 | 294.32 |
| Conv116 | 64 | 256 | 0.52 | 0.05 | 0.90 | 0.05 | 0.90 | 0.05 | 1.52 | 83.68 |
| Conv117 | 64 | 576 | 0.88 | 0.05 | 0.97 | 0.05 | 0.97 | 0.05 | 0.07 | 83.68 |
| Conv118 | 256 | 64 | 0.69 | 0.05 | 0.90 | 0.05 | 0.90 | 0.05 | 4.38 | 294.32 |
| Conv119 | 64 | 256 | 0.53 | 0.05 | 0.90 | 0.05 | 0.90 | 0.05 | 1.19 | 83.68 |
| Conv120 | 64 | 576 | 0.87 | 0.05 | 0.97 | 0.05 | 0.97 | 0.05 | 0.06 | 83.68 |
| Conv121 | 256 | 64 | 0.84 | 0.05 | 0.90 | 0.05 | 0.90 | 0.05 | 4.75 | 294.32 |
| Conv122 | 64 | 256 | 0.52 | 0.05 | 0.90 | 0.05 | 0.90 | 0.05 | 1.08 | 83.68 |
| Conv123 | 64 | 576 | 0.88 | 0.05 | 0.97 | 0.05 | 0.97 | 0.05 | 0.04 | 83.68 |
| Conv124 | 256 | 64 | 0.84 | 0.05 | 0.90 | 0.05 | 0.90 | 0.05 | 4.98 | 294.32 |
| Conv125 | 64 | 256 | 0.50 | 0.05 | 0.90 | 0.05 | 0.90 | 0.05 | 1.36 | 83.68 |
| Conv126 | 64 | 576 | 0.87 | 0.05 | 0.97 | 0.05 | 0.97 | 0.05 | 0.06 | 83.68 |
| Conv127 | 256 | 64 | 0.83 | 0.05 | 0.90 | 0.05 | 0.90 | 0.05 | 4.89 | 294.32 |
| Conv128 | 64 | 256 | 0.47 | 0.05 | 0.90 | 0.05 | 0.90 | 0.05 | 1.42 | 83.68 |
| Conv129 | 64 | 576 | 0.88 | 0.05 | 0.97 | 0.05 | 0.97 | 0.05 | 0.05 | 83.68 |
| Conv130 | 256 | 64 | 0.82 | 0.05 | 0.90 | 0.05 | 0.90 | 0.05 | 5.59 | 294.32 |
| Conv131 | 64 | 256 | 0.52 | 0.05 | 0.90 | 0.05 | 0.90 | 0.05 | 0.97 | 83.68 |
| Conv132 | 64 | 576 | 0.88 | 0.05 | 0.97 | 0.05 | 0.97 | 0.05 | 0.06 | 83.68 |
| Conv133 | 256 | 64 | 0.84 | 0.05 | 0.90 | 0.05 | 0.90 | 0.05 | 4.4 | 294.32 |
| Conv134 | 64 | 256 | 0.52 | 0.05 | 0.90 | 0.05 | 0.90 | 0.05 | 0.84 | 83.68 |
| Conv135 | 64 | 576 | 0.88 | 0.05 | 0.97 | 0.05 | 0.97 | 0.05 | 0.08 | 83.68 |
| Conv136 | 256 | 64 | 0.82 | 0.05 | 0.90 | 0.05 | 0.90 | 0.05 | 5.17 | 294.32 |
| Conv137 | 64 | 256 | 0.49 | 0.05 | 0.90 | 0.05 | 0.90 | 0.05 | 1.24 | 83.68 |
| Conv138 | 64 | 576 | 0.87 | 0.05 | 0.97 | 0.05 | 0.97 | 0.05 | 0.05 | 83.68 |
| Conv139 | 256 | 64 | 0.84 | 0.05 | 0.90 | 0.05 | 0.90 | 0.05 | 5.18 | 294.32 |
| Conv140 | 64 | 256 | 0.52 | 0.05 | 0.90 | 0.05 | 0.90 | 0.05 | 1.94 | 83.68 |
| Conv141 | 64 | 576 | 0.85 | 0.05 | 0.97 | 0.05 | 0.97 | 0.05 | 0.05 | 83.68 |
| Conv142 | 256 | 64 | 0.81 | 0.05 | 0.90 | 0.05 | 0.90 | 0.05 | 5.9 | 294.32 |
| Conv143 | 64 | 256 | 0.50 | 0.05 | 0.90 | 0.05 | 0.90 | 0.05 | 1.31 | 83.68 |
| Conv144 | 64 | 576 | 0.88 | 0.05 | 0.97 | 0.05 | 0.97 | 0.05 | 0.06 | 83.68 |
| Conv145 | 256 | 64 | 0.82 | 0.05 | 0.90 | 0.05 | 0.90 | 0.05 | 4.99 | 294.32 |
| Conv146 | 64 | 256 | 0.53 | 0.05 | 0.90 | 0.05 | 0.90 | 0.05 | 1.05 | 83.68 |
| Conv147 | 64 | 576 | 0.87 | 0.05 | 0.97 | 0.05 | 0.97 | 0.05 | 0.04 | 83.68 |
| Conv148 | 256 | 64 | 0.83 | 0.05 | 0.90 | 0.05 | 0.90 | 0.05 | 4.7 | 294.32 |
| Conv149 | 64 | 256 | 0.50 | 0.05 | 0.90 | 0.05 | 0.90 | 0.05 | 1.34 | 83.68 |
| Conv150 | 64 | 576 | 0.88 | 0.05 | 0.97 | 0.05 | 0.97 | 0.05 | 0.05 | 83.68 |
| Conv151 | 256 | 64 | 0.82 | 0.05 | 0.90 | 0.05 | 0.90 | 0.05 | 5.03 | 294.32 |
| Conv152 | 64 | 256 | 0.51 | 0.05 | 0.90 | 0.05 | 0.90 | 0.05 | 1.36 | 83.68 |
| Conv153 | 64 | 576 | 0.88 | 0.05 | 0.97 | 0.05 | 0.97 | 0.05 | 0.07 | 83.68 |
| Conv154 | 256 | 64 | 0.81 | 0.05 | 0.90 | 0.05 | 0.90 | 0.05 | 5.44 | 294.32 |
| Conv155 | 64 | 256 | 0.50 | 0.05 | 0.90 | 0.05 | 0.90 | 0.05 | 1.16 | 83.68 |
| Conv156 | 64 | 576 | 0.87 | 0.05 | 0.97 | 0.05 | 0.97 | 0.05 | 0.06 | 83.68 |
| Conv157 | 256 | 64 | 0.81 | 0.05 | 0.90 | 0.05 | 0.90 | 0.05 | 4.65 | 294.32 |
| Conv158 | 64 | 256 | 0.47 | 0.05 | 0.90 | 0.05 | 0.90 | 0.05 | 1.52 | 83.68 |
| Conv159 | 64 | 576 | 0.87 | 0.05 | 0.97 | 0.05 | 0.97 | 0.05 | 0.08 | 83.68 |
| Conv160 | 256 | 64 | 0.82 | 0.05 | 0.90 | 0.05 | 0.90 | 0.05 | 5.66 | 294.32 |
| Conv161 | 64 | 256 | 0.48 | 0.05 | 0.90 | 0.05 | 0.90 | 0.05 | 1.17 | 83.68 |
| Conv162 | 64 | 576 | 0.88 | 0.05 | 0.97 | 0.05 | 0.97 | 0.05 | 0.05 | 83.68 |
| Conv163 | 256 | 64 | 0.83 | 0.05 | 0.90 | 0.05 | 0.90 | 0.05 | 5.4 | 294.32 |
| **Passing rate** | - | - | 100.0% | | 100.0% | | 100.0% | | 99.39% | |

Table 16: Cifar100 Adam-ResNet164

| Layer | Number | dim | Gaussian | | Mean_Left | | Mean_Right | | Sigma | |
|-------|--------|-----|----------|--|-----------|--|------------|--|-------|--|

| | | | p-value | c-value | p-value | c-value | p-value | c-value | t-value | c-value |
|---|---|---|---|---|---|---|---|---|---|---|
| Conv1 | 16 | 27 | 0.03 | 0.05 | 0.58 | 0.05 | 0.58 | 0.05 | 466.72 | 26.3 |
| Conv2 | 16 | 16 | 0.12 | 0.05 | 0.56 | 0.05 | 0.56 | 0.05 | 222.95 | 26.3 |
| Conv3 | 16 | 144 | 0.43 | 0.05 | 0.68 | 0.05 | 0.68 | 0.05 | 31.17 | 26.3 |
| Conv4 | 64 | 16 | 0.35 | 0.05 | 0.63 | 0.05 | 0.63 | 0.05 | 103.14 | 83.68 |
| Conv5 | 16 | 64 | 0.34 | 0.05 | 0.63 | 0.05 | 0.63 | 0.05 | 17.8 | 26.3 |
| Conv6 | 16 | 144 | 0.52 | 0.05 | 0.68 | 0.05 | 0.68 | 0.05 | 5.97 | 26.3 |
| Conv7 | 64 | 16 | 0.69 | 0.05 | 0.63 | 0.05 | 0.63 | 0.05 | 12.77 | 83.68 |
| Conv8 | 16 | 64 | 0.33 | 0.05 | 0.63 | 0.05 | 0.63 | 0.05 | 31.39 | 26.3 |
| Conv9 | 16 | 144 | 0.41 | 0.05 | 0.68 | 0.05 | 0.68 | 0.05 | 16.75 | 26.3 |
| Conv10 | 64 | 16 | 0.66 | 0.05 | 0.63 | 0.05 | 0.63 | 0.05 | 26.55 | 83.68 |
| Conv11 | 16 | 64 | 0.50 | 0.05 | 0.63 | 0.05 | 0.63 | 0.05 | 8.63 | 26.3 |
| Conv12 | 16 | 144 | 0.61 | 0.05 | 0.68 | 0.05 | 0.68 | 0.05 | 3.88 | 26.3 |
| Conv13 | 64 | 16 | 0.67 | 0.05 | 0.63 | 0.05 | 0.63 | 0.05 | 22.11 | 83.68 |
| Conv14 | 16 | 64 | 0.66 | 0.05 | 0.63 | 0.05 | 0.63 | 0.05 | 5.01 | 26.3 |
| Conv15 | 16 | 144 | 0.69 | 0.05 | 0.68 | 0.05 | 0.68 | 0.05 | 2.1 | 26.3 |
| Conv16 | 64 | 16 | 0.87 | 0.05 | 0.63 | 0.05 | 0.63 | 0.05 | 5.14 | 83.68 |
| Conv17 | 16 | 64 | 0.28 | 0.05 | 0.63 | 0.05 | 0.63 | 0.05 | 20.05 | 26.3 |
| Conv18 | 16 | 144 | 0.17 | 0.05 | 0.68 | 0.05 | 0.68 | 0.05 | 7.17 | 26.3 |
| Conv19 | 64 | 16 | 0.86 | 0.05 | 0.63 | 0.05 | 0.63 | 0.05 | 4.1 | 83.68 |
| Conv20 | 16 | 64 | 0.57 | 0.05 | 0.63 | 0.05 | 0.63 | 0.05 | 17.79 | 26.3 |
| Conv21 | 16 | 144 | 0.31 | 0.05 | 0.68 | 0.05 | 0.68 | 0.05 | 9.78 | 26.3 |
| Conv22 | 64 | 16 | 0.91 | 0.05 | 0.63 | 0.05 | 0.63 | 0.05 | 3.07 | 83.68 |
| Conv23 | 16 | 64 | 0.42 | 0.05 | 0.63 | 0.05 | 0.63 | 0.05 | 21.74 | 26.3 |
| Conv24 | 16 | 144 | 0.31 | 0.05 | 0.68 | 0.05 | 0.68 | 0.05 | 10.79 | 26.3 |
| Conv25 | 64 | 16 | 0.65 | 0.05 | 0.63 | 0.05 | 0.63 | 0.05 | 13.76 | 83.68 |
| Conv26 | 16 | 64 | 0.34 | 0.05 | 0.63 | 0.05 | 0.63 | 0.05 | 14.28 | 26.3 |
| Conv27 | 16 | 144 | 0.29 | 0.05 | 0.68 | 0.05 | 0.68 | 0.05 | 13.23 | 26.3 |
| Conv28 | 64 | 16 | 0.54 | 0.05 | 0.63 | 0.05 | 0.63 | 0.05 | 31.67 | 83.68 |
| Conv29 | 16 | 64 | 0.33 | 0.05 | 0.63 | 0.05 | 0.63 | 0.05 | 14.97 | 26.3 |
| Conv30 | 16 | 144 | 0.50 | 0.05 | 0.68 | 0.05 | 0.68 | 0.05 | 5.76 | 26.3 |
| Conv31 | 64 | 16 | 0.46 | 0.05 | 0.63 | 0.05 | 0.63 | 0.05 | 25.54 | 83.68 |
| Conv32 | 16 | 64 | 0.39 | 0.05 | 0.63 | 0.05 | 0.63 | 0.05 | 14.78 | 26.3 |
| Conv33 | 16 | 144 | 0.38 | 0.05 | 0.68 | 0.05 | 0.68 | 0.05 | 6.19 | 26.3 |
| Conv34 | 64 | 16 | 0.84 | 0.05 | 0.63 | 0.05 | 0.63 | 0.05 | 9.25 | 83.68 |
| Conv35 | 16 | 64 | 0.47 | 0.05 | 0.63 | 0.05 | 0.63 | 0.05 | 11.5 | 26.3 |
| Conv36 | 16 | 144 | 0.42 | 0.05 | 0.68 | 0.05 | 0.68 | 0.05 | 13.26 | 26.3 |
| Conv37 | 64 | 16 | 0.55 | 0.05 | 0.63 | 0.05 | 0.63 | 0.05 | 77.76 | 83.68 |
| Conv38 | 16 | 64 | 0.46 | 0.05 | 0.63 | 0.05 | 0.63 | 0.05 | 20.83 | 26.3 |
| Conv39 | 16 | 144 | 0.38 | 0.05 | 0.68 | 0.05 | 0.68 | 0.05 | 11.74 | 26.3 |
| Conv40 | 64 | 16 | 0.68 | 0.05 | 0.63 | 0.05 | 0.63 | 0.05 | 35.12 | 83.68 |
| Conv41 | 16 | 64 | 0.92 | 0.05 | 0.63 | 0.05 | 0.63 | 0.05 | 0.53 | 26.3 |
| Conv42 | 16 | 144 | 0.94 | 0.05 | 0.68 | 0.05 | 0.68 | 0.05 | 0.15 | 26.3 |
| Conv43 | 64 | 16 | 0.94 | 0.05 | 0.63 | 0.05 | 0.63 | 0.05 | 1.15 | 83.68 |
| Conv44 | 16 | 64 | 0.42 | 0.05 | 0.63 | 0.05 | 0.63 | 0.05 | 20.83 | 26.3 |
| Conv45 | 16 | 144 | 0.46 | 0.05 | 0.68 | 0.05 | 0.68 | 0.05 | 15.2 | 26.3 |
| Conv46 | 64 | 16 | 0.45 | 0.05 | 0.63 | 0.05 | 0.63 | 0.05 | 50.71 | 83.68 |
| Conv47 | 16 | 64 | 0.30 | 0.05 | 0.63 | 0.05 | 0.63 | 0.05 | 25.92 | 26.3 |
| Conv48 | 16 | 144 | 0.43 | 0.05 | 0.68 | 0.05 | 0.68 | 0.05 | 15.15 | 26.3 |
| Conv49 | 64 | 16 | 0.53 | 0.05 | 0.63 | 0.05 | 0.63 | 0.05 | 77.85 | 83.68 |
| Conv50 | 16 | 64 | 0.34 | 0.05 | 0.63 | 0.05 | 0.63 | 0.05 | 23.91 | 26.3 |
| Conv51 | 16 | 144 | 0.52 | 0.05 | 0.68 | 0.05 | 0.68 | 0.05 | 12.91 | 26.3 |
| Conv52 | 64 | 16 | 0.45 | 0.05 | 0.63 | 0.05 | 0.63 | 0.05 | 74.94 | 83.68 |
| Conv53 | 16 | 64 | 0.41 | 0.05 | 0.63 | 0.05 | 0.63 | 0.05 | 30.44 | 26.3 |
| Conv54 | 16 | 144 | 0.41 | 0.05 | 0.68 | 0.05 | 0.68 | 0.05 | 21.99 | 26.3 |
| Conv55 | 64 | 16 | 0.41 | 0.05 | 0.63 | 0.05 | 0.63 | 0.05 | 106.69 | 83.68 |
| Conv56 | 32 | 64 | 0.13 | 0.05 | 0.67 | 0.05 | 0.67 | 0.05 | 167.36 | 46.19 |
| Conv57 | 32 | 288 | 0.42 | 0.05 | 0.83 | 0.05 | 0.83 | 0.05 | 28.09 | 46.19 |

| | | | | | | | | | | |
|---|---|---|---|---|---|---|---|---|---|---|
| Conv58 | 128 | 32 | 0.17 | 0.05 | 0.74 | 0.05 | 0.74 | 0.05 | 320.85 | 155.4 |
| Conv59 | 32 | 128 | 0.20 | 0.05 | 0.74 | 0.05 | 0.74 | 0.05 | 15.24 | 46.19 |
| Conv60 | 32 | 288 | 0.23 | 0.05 | 0.83 | 0.05 | 0.83 | 0.05 | 10.97 | 46.19 |
| Conv61 | 128 | 32 | 0.57 | 0.05 | 0.74 | 0.05 | 0.74 | 0.05 | 42.72 | 155.4 |
| Conv62 | 32 | 128 | 0.22 | 0.05 | 0.74 | 0.05 | 0.74 | 0.05 | 15.01 | 46.19 |
| Conv63 | 32 | 288 | 0.39 | 0.05 | 0.83 | 0.05 | 0.83 | 0.05 | 9.25 | 46.19 |
| Conv64 | 128 | 32 | 0.63 | 0.05 | 0.74 | 0.05 | 0.74 | 0.05 | 40.35 | 155.4 |
| Conv65 | 32 | 128 | 0.33 | 0.05 | 0.74 | 0.05 | 0.74 | 0.05 | 10.85 | 46.19 |
| Conv66 | 32 | 288 | 0.64 | 0.05 | 0.83 | 0.05 | 0.83 | 0.05 | 4.2 | 46.19 |
| Conv67 | 128 | 32 | 0.50 | 0.05 | 0.74 | 0.05 | 0.74 | 0.05 | 26.77 | 155.4 |
| Conv68 | 32 | 128 | 0.15 | 0.05 | 0.74 | 0.05 | 0.74 | 0.05 | 22.12 | 46.19 |
| Conv69 | 32 | 288 | 0.37 | 0.05 | 0.83 | 0.05 | 0.83 | 0.05 | 15.38 | 46.19 |
| Conv70 | 128 | 32 | 0.49 | 0.05 | 0.74 | 0.05 | 0.74 | 0.05 | 69.31 | 155.4 |
| Conv71 | 32 | 128 | 0.18 | 0.05 | 0.74 | 0.05 | 0.74 | 0.05 | 10.69 | 46.19 |
| Conv72 | 32 | 288 | 0.26 | 0.05 | 0.83 | 0.05 | 0.83 | 0.05 | 6.71 | 46.19 |
| Conv73 | 128 | 32 | 0.56 | 0.05 | 0.74 | 0.05 | 0.74 | 0.05 | 15.51 | 155.4 |
| Conv74 | 32 | 128 | 0.38 | 0.05 | 0.74 | 0.05 | 0.74 | 0.05 | 23.08 | 46.19 |
| Conv75 | 32 | 288 | 0.43 | 0.05 | 0.83 | 0.05 | 0.83 | 0.05 | 9.83 | 46.19 |
| Conv76 | 128 | 32 | 0.49 | 0.05 | 0.74 | 0.05 | 0.74 | 0.05 | 29.79 | 155.4 |
| Conv77 | 32 | 128 | 0.37 | 0.05 | 0.74 | 0.05 | 0.74 | 0.05 | 20.85 | 46.19 |
| Conv78 | 32 | 288 | 0.43 | 0.05 | 0.83 | 0.05 | 0.83 | 0.05 | 7.53 | 46.19 |
| Conv79 | 128 | 32 | 0.58 | 0.05 | 0.74 | 0.05 | 0.74 | 0.05 | 38.54 | 155.4 |
| Conv80 | 32 | 128 | 0.26 | 0.05 | 0.74 | 0.05 | 0.74 | 0.05 | 18.75 | 46.19 |
| Conv81 | 32 | 288 | 0.38 | 0.05 | 0.83 | 0.05 | 0.83 | 0.05 | 12.27 | 46.19 |
| Conv82 | 128 | 32 | 0.62 | 0.05 | 0.74 | 0.05 | 0.74 | 0.05 | 32.84 | 155.4 |
| Conv83 | 32 | 128 | 0.10 | 0.05 | 0.74 | 0.05 | 0.74 | 0.05 | 20.59 | 46.19 |
| Conv84 | 32 | 288 | 0.31 | 0.05 | 0.83 | 0.05 | 0.83 | 0.05 | 16.6 | 46.19 |
| Conv85 | 128 | 32 | 0.38 | 0.05 | 0.74 | 0.05 | 0.74 | 0.05 | 107.99 | 155.4 |
| Conv86 | 32 | 128 | 0.44 | 0.05 | 0.74 | 0.05 | 0.74 | 0.05 | 18.28 | 46.19 |
| Conv87 | 32 | 288 | 0.47 | 0.05 | 0.83 | 0.05 | 0.83 | 0.05 | 10.62 | 46.19 |
| Conv88 | 128 | 32 | 0.62 | 0.05 | 0.74 | 0.05 | 0.74 | 0.05 | 35.83 | 155.4 |
| Conv89 | 32 | 128 | 0.21 | 0.05 | 0.74 | 0.05 | 0.74 | 0.05 | 21.55 | 46.19 |
| Conv90 | 32 | 288 | 0.57 | 0.05 | 0.83 | 0.05 | 0.83 | 0.05 | 7.98 | 46.19 |
| Conv91 | 128 | 32 | 0.49 | 0.05 | 0.74 | 0.05 | 0.74 | 0.05 | 66.43 | 155.4 |
| Conv92 | 32 | 128 | 0.43 | 0.05 | 0.74 | 0.05 | 0.74 | 0.05 | 17.04 | 46.19 |
| Conv93 | 32 | 288 | 0.30 | 0.05 | 0.83 | 0.05 | 0.83 | 0.05 | 8.51 | 46.19 |
| Conv94 | 128 | 32 | 0.58 | 0.05 | 0.74 | 0.05 | 0.74 | 0.05 | 43.25 | 155.4 |
| Conv95 | 32 | 128 | 0.19 | 0.05 | 0.74 | 0.05 | 0.74 | 0.05 | 14.97 | 46.19 |
| Conv96 | 32 | 288 | 0.29 | 0.05 | 0.83 | 0.05 | 0.83 | 0.05 | 9.6 | 46.19 |
| Conv97 | 128 | 32 | 0.60 | 0.05 | 0.74 | 0.05 | 0.74 | 0.05 | 36.05 | 155.4 |
| Conv98 | 32 | 128 | 0.35 | 0.05 | 0.74 | 0.05 | 0.74 | 0.05 | 7.96 | 46.19 |
| Conv99 | 32 | 288 | 0.46 | 0.05 | 0.83 | 0.05 | 0.83 | 0.05 | 2.19 | 46.19 |
| Conv100 | 128 | 32 | 0.81 | 0.05 | 0.74 | 0.05 | 0.74 | 0.05 | 11.53 | 155.4 |
| Conv101 | 32 | 128 | 0.33 | 0.05 | 0.74 | 0.05 | 0.74 | 0.05 | 16.35 | 46.19 |
| Conv102 | 32 | 288 | 0.34 | 0.05 | 0.83 | 0.05 | 0.83 | 0.05 | 9.13 | 46.19 |
| Conv103 | 128 | 32 | 0.59 | 0.05 | 0.74 | 0.05 | 0.74 | 0.05 | 21.11 | 155.4 |
| Conv104 | 32 | 128 | 0.39 | 0.05 | 0.74 | 0.05 | 0.74 | 0.05 | 20.91 | 46.19 |
| Conv105 | 32 | 288 | 0.64 | 0.05 | 0.83 | 0.05 | 0.83 | 0.05 | 8.36 | 46.19 |
| Conv106 | 128 | 32 | 0.54 | 0.05 | 0.74 | 0.05 | 0.74 | 0.05 | 44.06 | 155.4 |
| Conv107 | 32 | 128 | 0.31 | 0.05 | 0.74 | 0.05 | 0.74 | 0.05 | 29.98 | 46.19 |
| Conv108 | 32 | 288 | 0.45 | 0.05 | 0.83 | 0.05 | 0.83 | 0.05 | 15.03 | 46.19 |
| Conv109 | 128 | 32 | 0.50 | 0.05 | 0.74 | 0.05 | 0.74 | 0.05 | 82.36 | 155.4 |
| Conv110 | 64 | 128 | 0.14 | 0.05 | 0.82 | 0.05 | 0.82 | 0.05 | 197.08 | 83.68 |
| Conv111 | 64 | 576 | 0.46 | 0.05 | 0.97 | 0.05 | 0.97 | 0.05 | 59.29 | 83.68 |
| Conv112 | 256 | 64 | 0.16 | 0.05 | 0.90 | 0.05 | 0.90 | 0.05 | 594.89 | 294.32 |
| Conv113 | 64 | 256 | 0.44 | 0.05 | 0.90 | 0.05 | 0.90 | 0.05 | 33.77 | 83.68 |
| Conv114 | 64 | 576 | 0.32 | 0.05 | 0.97 | 0.05 | 0.97 | 0.05 | 26.61 | 83.68 |
| Conv115 | 256 | 64 | 0.30 | 0.05 | 0.90 | 0.05 | 0.90 | 0.05 | 175.45 | 294.32 |
| Conv116 | 64 | 256 | 0.38 | 0.05 | 0.90 | 0.05 | 0.90 | 0.05 | 24.35 | 83.68 |

| Layer | Number | dim | p-value | c-value | p-value | c-value | p-value | c-value | t-value | c-value |
|---|---|---|---|---|---|---|---|---|---|---|
| Conv117 | 64 | 576 | 0.33 | 0.05 | 0.97 | 0.05 | 0.97 | 0.05 | 16.49 | 83.68 |
| Conv118 | 256 | 64 | 0.51 | 0.05 | 0.90 | 0.05 | 0.90 | 0.05 | 123.0 | 294.32 |
| Conv119 | 64 | 256 | 0.08 | 0.05 | 0.90 | 0.05 | 0.90 | 0.05 | 35.01 | 83.68 |
| Conv120 | 64 | 576 | 0.34 | 0.05 | 0.97 | 0.05 | 0.97 | 0.05 | 24.5 | 83.68 |
| Conv121 | 256 | 64 | 0.36 | 0.05 | 0.90 | 0.05 | 0.90 | 0.05 | 159.72 | 294.32 |
| Conv122 | 64 | 256 | 0.42 | 0.05 | 0.90 | 0.05 | 0.90 | 0.05 | 21.93 | 83.68 |
| Conv123 | 64 | 576 | 0.16 | 0.05 | 0.97 | 0.05 | 0.97 | 0.05 | 12.24 | 83.68 |
| Conv124 | 256 | 64 | 0.54 | 0.05 | 0.90 | 0.05 | 0.90 | 0.05 | 86.27 | 294.32 |
| Conv125 | 64 | 256 | 0.24 | 0.05 | 0.90 | 0.05 | 0.90 | 0.05 | 38.93 | 83.68 |
| Conv126 | 64 | 576 | 0.44 | 0.05 | 0.97 | 0.05 | 0.97 | 0.05 | 17.46 | 83.68 |
| Conv127 | 256 | 64 | 0.50 | 0.05 | 0.90 | 0.05 | 0.90 | 0.05 | 91.81 | 294.32 |
| Conv128 | 64 | 256 | 0.32 | 0.05 | 0.90 | 0.05 | 0.90 | 0.05 | 29.94 | 83.68 |
| Conv129 | 64 | 576 | 0.33 | 0.05 | 0.97 | 0.05 | 0.97 | 0.05 | 11.34 | 83.68 |
| Conv130 | 256 | 64 | 0.58 | 0.05 | 0.90 | 0.05 | 0.90 | 0.05 | 76.05 | 294.32 |
| Conv131 | 64 | 256 | 0.46 | 0.05 | 0.90 | 0.05 | 0.90 | 0.05 | 38.51 | 83.68 |
| Conv132 | 64 | 576 | 0.42 | 0.05 | 0.97 | 0.05 | 0.97 | 0.05 | 13.99 | 83.68 |
| Conv133 | 256 | 64 | 0.49 | 0.05 | 0.90 | 0.05 | 0.90 | 0.05 | 78.35 | 294.32 |
| Conv134 | 64 | 256 | 0.28 | 0.05 | 0.90 | 0.05 | 0.90 | 0.05 | 55.05 | 83.68 |
| Conv135 | 64 | 576 | 0.34 | 0.05 | 0.97 | 0.05 | 0.97 | 0.05 | 22.32 | 83.68 |
| Conv136 | 256 | 64 | 0.40 | 0.05 | 0.90 | 0.05 | 0.90 | 0.05 | 127.27 | 294.32 |
| Conv137 | 64 | 256 | 0.45 | 0.05 | 0.90 | 0.05 | 0.90 | 0.05 | 73.92 | 83.68 |
| Conv138 | 64 | 576 | 0.39 | 0.05 | 0.97 | 0.05 | 0.97 | 0.05 | 20.65 | 83.68 |
| Conv139 | 256 | 64 | 0.43 | 0.05 | 0.90 | 0.05 | 0.90 | 0.05 | 79.88 | 294.32 |
| Conv140 | 64 | 256 | 0.46 | 0.05 | 0.90 | 0.05 | 0.90 | 0.05 | 57.87 | 83.68 |
| Conv141 | 64 | 576 | 0.46 | 0.05 | 0.97 | 0.05 | 0.97 | 0.05 | 14.7 | 83.68 |
| Conv142 | 256 | 64 | 0.49 | 0.05 | 0.90 | 0.05 | 0.90 | 0.05 | 62.55 | 294.32 |
| Conv143 | 64 | 256 | 0.36 | 0.05 | 0.90 | 0.05 | 0.90 | 0.05 | 64.03 | 83.68 |
| Conv144 | 64 | 576 | 0.51 | 0.05 | 0.97 | 0.05 | 0.97 | 0.05 | 17.48 | 83.68 |
| Conv145 | 256 | 64 | 0.45 | 0.05 | 0.90 | 0.05 | 0.90 | 0.05 | 59.58 | 294.32 |
| Conv146 | 64 | 256 | 0.36 | 0.05 | 0.90 | 0.05 | 0.90 | 0.05 | 106.22 | 83.68 |
| Conv147 | 64 | 576 | 0.53 | 0.05 | 0.97 | 0.05 | 0.97 | 0.05 | 32.45 | 83.68 |
| Conv148 | 256 | 64 | 0.39 | 0.05 | 0.90 | 0.05 | 0.90 | 0.05 | 110.05 | 294.32 |
| Conv149 | 64 | 256 | 0.43 | 0.05 | 0.90 | 0.05 | 0.90 | 0.05 | 105.01 | 83.68 |
| Conv150 | 64 | 576 | 0.43 | 0.05 | 0.97 | 0.05 | 0.97 | 0.05 | 30.32 | 83.68 |
| Conv151 | 256 | 64 | 0.40 | 0.05 | 0.90 | 0.05 | 0.90 | 0.05 | 81.71 | 294.32 |
| Conv152 | 64 | 256 | 0.34 | 0.05 | 0.90 | 0.05 | 0.90 | 0.05 | 113.29 | 83.68 |
| Conv153 | 64 | 576 | 0.42 | 0.05 | 0.97 | 0.05 | 0.97 | 0.05 | 23.49 | 83.68 |
| Conv154 | 256 | 64 | 0.37 | 0.05 | 0.90 | 0.05 | 0.90 | 0.05 | 75.03 | 294.32 |
| Conv155 | 64 | 256 | 0.30 | 0.05 | 0.90 | 0.05 | 0.90 | 0.05 | 101.39 | 83.68 |
| Conv156 | 64 | 576 | 0.63 | 0.05 | 0.97 | 0.05 | 0.97 | 0.05 | 15.84 | 83.68 |
| Conv157 | 256 | 64 | 0.49 | 0.05 | 0.90 | 0.05 | 0.90 | 0.05 | 58.98 | 294.32 |
| Conv158 | 64 | 256 | 0.23 | 0.05 | 0.90 | 0.05 | 0.90 | 0.05 | 84.02 | 83.68 |
| Conv159 | 64 | 576 | 0.65 | 0.05 | 0.97 | 0.05 | 0.97 | 0.05 | 8.42 | 83.68 |
| Conv160 | 256 | 64 | 0.54 | 0.05 | 0.90 | 0.05 | 0.90 | 0.05 | 33.29 | 294.32 |
| Conv161 | 64 | 256 | 0.57 | 0.05 | 0.90 | 0.05 | 0.90 | 0.05 | 20.57 | 83.68 |
| Conv162 | 64 | 576 | 0.82 | 0.05 | 0.97 | 0.05 | 0.97 | 0.05 | 1.41 | 83.68 |
| Conv163 | 256 | 64 | 0.86 | 0.05 | 0.90 | 0.05 | 0.90 | 0.05 | 9.3 | 294.32 |
| **Passing rate** | - | - | 99.39% | | 100.0% | | 100.0% | | 90.18% | |

Table 17: Cifar100 Adagrad-ResNet164

| Layer | Number | dim | Gaussian | | Mean_Left | | Mean_Right | | Sigma | |
|---|---|---|---|---|---|---|---|---|---|---|
| | | | p-value | c-value | p-value | c-value | p-value | c-value | t-value | c-value |
| Conv1 | 16 | 27 | 0.81 | 0.05 | 0.58 | 0.05 | 0.58 | 0.05 | 1.0 | 26.3 |
| Conv2 | 16 | 16 | 0.29 | 0.05 | 0.56 | 0.05 | 0.56 | 0.05 | 27.5 | 26.3 |

| | | | | | | | | | | | |
|---|---|---|---|---|---|---|---|---|---|---|---|
| Conv3 | 16 | 144 | 0.64 | 0.05 | 0.68 | 0.05 | 0.68 | 0.05 | 0.21 | 26.3 |
| Conv4 | 64 | 16 | 0.58 | 0.05 | 0.63 | 0.05 | 0.63 | 0.05 | 26.43 | 83.68 |
| Conv5 | 16 | 64 | 0.22 | 0.05 | 0.63 | 0.05 | 0.63 | 0.05 | 5.73 | 26.3 |
| Conv6 | 16 | 144 | 0.69 | 0.05 | 0.68 | 0.05 | 0.68 | 0.05 | 0.31 | 26.3 |
| Conv7 | 64 | 16 | 0.56 | 0.05 | 0.63 | 0.05 | 0.63 | 0.05 | 24.75 | 83.68 |
| Conv8 | 16 | 64 | 0.26 | 0.05 | 0.63 | 0.05 | 0.63 | 0.05 | 2.68 | 26.3 |
| Conv9 | 16 | 144 | 0.67 | 0.05 | 0.68 | 0.05 | 0.68 | 0.05 | 0.32 | 26.3 |
| Conv10 | 64 | 16 | 0.57 | 0.05 | 0.63 | 0.05 | 0.63 | 0.05 | 23.47 | 83.68 |
| Conv11 | 16 | 64 | 0.32 | 0.05 | 0.63 | 0.05 | 0.63 | 0.05 | 4.14 | 26.3 |
| Conv12 | 16 | 144 | 0.68 | 0.05 | 0.68 | 0.05 | 0.68 | 0.05 | 0.16 | 26.3 |
| Conv13 | 64 | 16 | 0.49 | 0.05 | 0.63 | 0.05 | 0.63 | 0.05 | 21.49 | 83.68 |
| Conv14 | 16 | 64 | 0.25 | 0.05 | 0.63 | 0.05 | 0.63 | 0.05 | 7.46 | 26.3 |
| Conv15 | 16 | 144 | 0.69 | 0.05 | 0.68 | 0.05 | 0.68 | 0.05 | 0.22 | 26.3 |
| Conv16 | 64 | 16 | 0.56 | 0.05 | 0.63 | 0.05 | 0.63 | 0.05 | 22.21 | 83.68 |
| Conv17 | 16 | 64 | 0.24 | 0.05 | 0.63 | 0.05 | 0.63 | 0.05 | 3.57 | 26.3 |
| Conv18 | 16 | 144 | 0.68 | 0.05 | 0.68 | 0.05 | 0.68 | 0.05 | 0.36 | 26.3 |
| Conv19 | 64 | 16 | 0.57 | 0.05 | 0.63 | 0.05 | 0.63 | 0.05 | 27.9 | 83.68 |
| Conv20 | 16 | 64 | 0.22 | 0.05 | 0.63 | 0.05 | 0.63 | 0.05 | 3.02 | 26.3 |
| Conv21 | 16 | 144 | 0.64 | 0.05 | 0.68 | 0.05 | 0.68 | 0.05 | 0.18 | 26.3 |
| Conv22 | 64 | 16 | 0.57 | 0.05 | 0.63 | 0.05 | 0.63 | 0.05 | 18.51 | 83.68 |
| Conv23 | 16 | 64 | 0.27 | 0.05 | 0.63 | 0.05 | 0.63 | 0.05 | 3.4 | 26.3 |
| Conv24 | 16 | 144 | 0.71 | 0.05 | 0.68 | 0.05 | 0.68 | 0.05 | 0.24 | 26.3 |
| Conv25 | 64 | 16 | 0.58 | 0.05 | 0.63 | 0.05 | 0.63 | 0.05 | 15.57 | 83.68 |
| Conv26 | 16 | 64 | 0.30 | 0.05 | 0.63 | 0.05 | 0.63 | 0.05 | 3.65 | 26.3 |
| Conv27 | 16 | 144 | 0.71 | 0.05 | 0.68 | 0.05 | 0.68 | 0.05 | 0.24 | 26.3 |
| Conv28 | 64 | 16 | 0.51 | 0.05 | 0.63 | 0.05 | 0.63 | 0.05 | 20.17 | 83.68 |
| Conv29 | 16 | 64 | 0.21 | 0.05 | 0.63 | 0.05 | 0.63 | 0.05 | 5.1 | 26.3 |
| Conv30 | 16 | 144 | 0.69 | 0.05 | 0.68 | 0.05 | 0.68 | 0.05 | 0.23 | 26.3 |
| Conv31 | 64 | 16 | 0.54 | 0.05 | 0.63 | 0.05 | 0.63 | 0.05 | 21.99 | 83.68 |
| Conv32 | 16 | 64 | 0.30 | 0.05 | 0.63 | 0.05 | 0.63 | 0.05 | 5.37 | 26.3 |
| Conv33 | 16 | 144 | 0.65 | 0.05 | 0.68 | 0.05 | 0.68 | 0.05 | 0.41 | 26.3 |
| Conv34 | 64 | 16 | 0.49 | 0.05 | 0.63 | 0.05 | 0.63 | 0.05 | 22.39 | 83.68 |
| Conv35 | 16 | 64 | 0.20 | 0.05 | 0.63 | 0.05 | 0.63 | 0.05 | 5.39 | 26.3 |
| Conv36 | 16 | 144 | 0.63 | 0.05 | 0.68 | 0.05 | 0.68 | 0.05 | 0.15 | 26.3 |
| Conv37 | 64 | 16 | 0.60 | 0.05 | 0.63 | 0.05 | 0.63 | 0.05 | 22.47 | 83.68 |
| Conv38 | 16 | 64 | 0.33 | 0.05 | 0.63 | 0.05 | 0.63 | 0.05 | 2.65 | 26.3 |
| Conv39 | 16 | 144 | 0.68 | 0.05 | 0.68 | 0.05 | 0.68 | 0.05 | 0.22 | 26.3 |
| Conv40 | 64 | 16 | 0.56 | 0.05 | 0.63 | 0.05 | 0.63 | 0.05 | 15.34 | 83.68 |
| Conv41 | 16 | 64 | 0.34 | 0.05 | 0.63 | 0.05 | 0.63 | 0.05 | 4.43 | 26.3 |
| Conv42 | 16 | 144 | 0.69 | 0.05 | 0.68 | 0.05 | 0.68 | 0.05 | 0.28 | 26.3 |
| Conv43 | 64 | 16 | 0.61 | 0.05 | 0.63 | 0.05 | 0.63 | 0.05 | 20.54 | 83.68 |
| Conv44 | 16 | 64 | 0.34 | 0.05 | 0.63 | 0.05 | 0.63 | 0.05 | 5.02 | 26.3 |
| Conv45 | 16 | 144 | 0.67 | 0.05 | 0.68 | 0.05 | 0.68 | 0.05 | 0.37 | 26.3 |
| Conv46 | 64 | 16 | 0.63 | 0.05 | 0.63 | 0.05 | 0.63 | 0.05 | 19.02 | 83.68 |
| Conv47 | 16 | 64 | 0.28 | 0.05 | 0.63 | 0.05 | 0.63 | 0.05 | 5.11 | 26.3 |
| Conv48 | 16 | 144 | 0.68 | 0.05 | 0.68 | 0.05 | 0.68 | 0.05 | 0.29 | 26.3 |
| Conv49 | 64 | 16 | 0.65 | 0.05 | 0.63 | 0.05 | 0.63 | 0.05 | 19.48 | 83.68 |
| Conv50 | 16 | 64 | 0.36 | 0.05 | 0.63 | 0.05 | 0.63 | 0.05 | 3.04 | 26.3 |
| Conv51 | 16 | 144 | 0.69 | 0.05 | 0.68 | 0.05 | 0.68 | 0.05 | 0.41 | 26.3 |
| Conv52 | 64 | 16 | 0.64 | 0.05 | 0.63 | 0.05 | 0.63 | 0.05 | 21.89 | 83.68 |
| Conv53 | 16 | 64 | 0.26 | 0.05 | 0.63 | 0.05 | 0.63 | 0.05 | 3.49 | 26.3 |
| Conv54 | 16 | 144 | 0.70 | 0.05 | 0.68 | 0.05 | 0.68 | 0.05 | 0.07 | 26.3 |
| Conv55 | 64 | 16 | 0.59 | 0.05 | 0.63 | 0.05 | 0.63 | 0.05 | 20.2 | 83.68 |
| Conv56 | 32 | 64 | 0.47 | 0.05 | 0.67 | 0.05 | 0.67 | 0.05 | 3.86 | 46.19 |
| Conv57 | 32 | 288 | 0.85 | 0.05 | 0.83 | 0.05 | 0.83 | 0.05 | 0.12 | 46.19 |
| Conv58 | 128 | 32 | 0.65 | 0.05 | 0.74 | 0.05 | 0.74 | 0.05 | 10.75 | 155.4 |
| Conv59 | 32 | 128 | 0.48 | 0.05 | 0.74 | 0.05 | 0.74 | 0.05 | 1.97 | 46.19 |
| Conv60 | 32 | 288 | 0.82 | 0.05 | 0.83 | 0.05 | 0.83 | 0.05 | 0.14 | 46.19 |
| Conv61 | 128 | 32 | 0.64 | 0.05 | 0.74 | 0.05 | 0.74 | 0.05 | 10.13 | 155.4 |

| | | | | | | | | | | |
|---|---|---|---|---|---|---|---|---|---|---|
| Conv62 | 32 | 128 | 0.48 | 0.05 | 0.74 | 0.05 | 0.74 | 0.05 | 2.12 | 46.19 |
| Conv63 | 32 | 288 | 0.83 | 0.05 | 0.83 | 0.05 | 0.83 | 0.05 | 0.21 | 46.19 |
| Conv64 | 128 | 32 | 0.67 | 0.05 | 0.74 | 0.05 | 0.74 | 0.05 | 9.34 | 155.4 |
| Conv65 | 32 | 128 | 0.48 | 0.05 | 0.74 | 0.05 | 0.74 | 0.05 | 4.38 | 46.19 |
| Conv66 | 32 | 288 | 0.85 | 0.05 | 0.83 | 0.05 | 0.83 | 0.05 | 0.23 | 46.19 |
| Conv67 | 128 | 32 | 0.65 | 0.05 | 0.74 | 0.05 | 0.74 | 0.05 | 9.64 | 155.4 |
| Conv68 | 32 | 128 | 0.43 | 0.05 | 0.74 | 0.05 | 0.74 | 0.05 | 4.33 | 46.19 |
| Conv69 | 32 | 288 | 0.84 | 0.05 | 0.83 | 0.05 | 0.83 | 0.05 | 0.22 | 46.19 |
| Conv70 | 128 | 32 | 0.68 | 0.05 | 0.74 | 0.05 | 0.74 | 0.05 | 9.63 | 155.4 |
| Conv71 | 32 | 128 | 0.48 | 0.05 | 0.74 | 0.05 | 0.74 | 0.05 | 3.55 | 46.19 |
| Conv72 | 32 | 288 | 0.84 | 0.05 | 0.83 | 0.05 | 0.83 | 0.05 | 0.19 | 46.19 |
| Conv73 | 128 | 32 | 0.81 | 0.05 | 0.74 | 0.05 | 0.74 | 0.05 | 9.45 | 155.4 |
| Conv74 | 32 | 128 | 0.44 | 0.05 | 0.74 | 0.05 | 0.74 | 0.05 | 5.7 | 46.19 |
| Conv75 | 32 | 288 | 0.84 | 0.05 | 0.83 | 0.05 | 0.83 | 0.05 | 0.41 | 46.19 |
| Conv76 | 128 | 32 | 0.69 | 0.05 | 0.74 | 0.05 | 0.74 | 0.05 | 10.14 | 155.4 |
| Conv77 | 32 | 128 | 0.39 | 0.05 | 0.74 | 0.05 | 0.74 | 0.05 | 6.42 | 46.19 |
| Conv78 | 32 | 288 | 0.83 | 0.05 | 0.83 | 0.05 | 0.83 | 0.05 | 0.25 | 46.19 |
| Conv79 | 128 | 32 | 0.67 | 0.05 | 0.74 | 0.05 | 0.74 | 0.05 | 7.54 | 155.4 |
| Conv80 | 32 | 128 | 0.52 | 0.05 | 0.74 | 0.05 | 0.74 | 0.05 | 3.95 | 46.19 |
| Conv81 | 32 | 288 | 0.85 | 0.05 | 0.83 | 0.05 | 0.83 | 0.05 | 0.24 | 46.19 |
| Conv82 | 128 | 32 | 0.70 | 0.05 | 0.74 | 0.05 | 0.74 | 0.05 | 8.76 | 155.4 |
| Conv83 | 32 | 128 | 0.53 | 0.05 | 0.74 | 0.05 | 0.74 | 0.05 | 6.0 | 46.19 |
| Conv84 | 32 | 288 | 0.85 | 0.05 | 0.83 | 0.05 | 0.83 | 0.05 | 0.33 | 46.19 |
| Conv85 | 128 | 32 | 0.70 | 0.05 | 0.74 | 0.05 | 0.74 | 0.05 | 9.76 | 155.4 |
| Conv86 | 32 | 128 | 0.51 | 0.05 | 0.74 | 0.05 | 0.74 | 0.05 | 3.95 | 46.19 |
| Conv87 | 32 | 288 | 0.85 | 0.05 | 0.83 | 0.05 | 0.83 | 0.05 | 0.39 | 46.19 |
| Conv88 | 128 | 32 | 0.81 | 0.05 | 0.74 | 0.05 | 0.74 | 0.05 | 10.14 | 155.4 |
| Conv89 | 32 | 128 | 0.54 | 0.05 | 0.74 | 0.05 | 0.74 | 0.05 | 6.26 | 46.19 |
| Conv90 | 32 | 288 | 0.86 | 0.05 | 0.83 | 0.05 | 0.83 | 0.05 | 0.62 | 46.19 |
| Conv91 | 128 | 32 | 0.68 | 0.05 | 0.74 | 0.05 | 0.74 | 0.05 | 10.09 | 155.4 |
| Conv92 | 32 | 128 | 0.48 | 0.05 | 0.74 | 0.05 | 0.74 | 0.05 | 3.63 | 46.19 |
| Conv93 | 32 | 288 | 0.86 | 0.05 | 0.83 | 0.05 | 0.83 | 0.05 | 0.33 | 46.19 |
| Conv94 | 128 | 32 | 0.71 | 0.05 | 0.74 | 0.05 | 0.74 | 0.05 | 9.94 | 155.4 |
| Conv95 | 32 | 128 | 0.53 | 0.05 | 0.74 | 0.05 | 0.74 | 0.05 | 3.39 | 46.19 |
| Conv96 | 32 | 288 | 0.87 | 0.05 | 0.83 | 0.05 | 0.83 | 0.05 | 0.4 | 46.19 |
| Conv97 | 128 | 32 | 0.69 | 0.05 | 0.74 | 0.05 | 0.74 | 0.05 | 6.74 | 155.4 |
| Conv98 | 32 | 128 | 0.52 | 0.05 | 0.74 | 0.05 | 0.74 | 0.05 | 7.89 | 46.19 |
| Conv99 | 32 | 288 | 0.85 | 0.05 | 0.83 | 0.05 | 0.83 | 0.05 | 0.38 | 46.19 |
| Conv100 | 128 | 32 | 0.82 | 0.05 | 0.74 | 0.05 | 0.74 | 0.05 | 9.47 | 155.4 |
| Conv101 | 32 | 128 | 0.55 | 0.05 | 0.74 | 0.05 | 0.74 | 0.05 | 4.92 | 46.19 |
| Conv102 | 32 | 288 | 0.85 | 0.05 | 0.83 | 0.05 | 0.83 | 0.05 | 0.37 | 46.19 |
| Conv103 | 128 | 32 | 0.82 | 0.05 | 0.74 | 0.05 | 0.74 | 0.05 | 10.66 | 155.4 |
| Conv104 | 32 | 128 | 0.49 | 0.05 | 0.74 | 0.05 | 0.74 | 0.05 | 4.22 | 46.19 |
| Conv105 | 32 | 288 | 0.84 | 0.05 | 0.83 | 0.05 | 0.83 | 0.05 | 0.34 | 46.19 |
| Conv106 | 128 | 32 | 0.81 | 0.05 | 0.74 | 0.05 | 0.74 | 0.05 | 9.11 | 155.4 |
| Conv107 | 32 | 128 | 0.51 | 0.05 | 0.74 | 0.05 | 0.74 | 0.05 | 5.88 | 46.19 |
| Conv108 | 32 | 288 | 0.85 | 0.05 | 0.83 | 0.05 | 0.83 | 0.05 | 0.42 | 46.19 |
| Conv109 | 128 | 32 | 0.71 | 0.05 | 0.74 | 0.05 | 0.74 | 0.05 | 7.11 | 155.4 |
| Conv110 | 64 | 128 | 0.55 | 0.05 | 0.82 | 0.05 | 0.82 | 0.05 | 3.26 | 83.68 |
| Conv111 | 64 | 576 | 0.87 | 0.05 | 0.97 | 0.05 | 0.97 | 0.05 | 0.2 | 83.68 |
| Conv112 | 256 | 64 | 0.83 | 0.05 | 0.90 | 0.05 | 0.90 | 0.05 | 5.43 | 294.32 |
| Conv113 | 64 | 256 | 0.63 | 0.05 | 0.90 | 0.05 | 0.90 | 0.05 | 2.29 | 83.68 |
| Conv114 | 64 | 576 | 0.88 | 0.05 | 0.97 | 0.05 | 0.97 | 0.05 | 0.23 | 83.68 |
| Conv115 | 256 | 64 | 0.83 | 0.05 | 0.90 | 0.05 | 0.90 | 0.05 | 6.3 | 294.32 |
| Conv116 | 64 | 256 | 0.62 | 0.05 | 0.90 | 0.05 | 0.90 | 0.05 | 2.39 | 83.68 |
| Conv117 | 64 | 576 | 0.89 | 0.05 | 0.97 | 0.05 | 0.97 | 0.05 | 0.25 | 83.68 |
| Conv118 | 256 | 64 | 0.86 | 0.05 | 0.90 | 0.05 | 0.90 | 0.05 | 6.52 | 294.32 |
| Conv119 | 64 | 256 | 0.60 | 0.05 | 0.90 | 0.05 | 0.90 | 0.05 | 3.15 | 83.68 |
| Conv120 | 64 | 576 | 0.89 | 0.05 | 0.97 | 0.05 | 0.97 | 0.05 | 0.28 | 83.68 |

| Layer | Number | dim | | | | | | | | |
|-------|--------|-----|------|------|------|------|------|------|------|------|
| Conv121 | 256 | 64 | 0.86 | 0.05 | 0.90 | 0.05 | 0.90 | 0.05 | 6.03 | 294.32 |
| Conv122 | 64 | 256 | 0.61 | 0.05 | 0.90 | 0.05 | 0.90 | 0.05 | 4.13 | 83.68 |
| Conv123 | 64 | 576 | 0.86 | 0.05 | 0.97 | 0.05 | 0.97 | 0.05 | 0.22 | 83.68 |
| Conv124 | 256 | 64 | 0.85 | 0.05 | 0.90 | 0.05 | 0.90 | 0.05 | 5.46 | 294.32 |
| Conv125 | 64 | 256 | 0.53 | 0.05 | 0.90 | 0.05 | 0.90 | 0.05 | 5.63 | 83.68 |
| Conv126 | 64 | 576 | 0.88 | 0.05 | 0.97 | 0.05 | 0.97 | 0.05 | 0.21 | 83.68 |
| Conv127 | 256 | 64 | 0.85 | 0.05 | 0.90 | 0.05 | 0.90 | 0.05 | 6.59 | 294.32 |
| Conv128 | 64 | 256 | 0.63 | 0.05 | 0.90 | 0.05 | 0.90 | 0.05 | 4.57 | 83.68 |
| Conv129 | 64 | 576 | 0.88 | 0.05 | 0.97 | 0.05 | 0.97 | 0.05 | 0.21 | 83.68 |
| Conv130 | 256 | 64 | 0.85 | 0.05 | 0.90 | 0.05 | 0.90 | 0.05 | 5.21 | 294.32 |
| Conv131 | 64 | 256 | 0.68 | 0.05 | 0.90 | 0.05 | 0.90 | 0.05 | 4.42 | 83.68 |
| Conv132 | 64 | 576 | 0.88 | 0.05 | 0.97 | 0.05 | 0.97 | 0.05 | 0.21 | 83.68 |
| Conv133 | 256 | 64 | 0.85 | 0.05 | 0.90 | 0.05 | 0.90 | 0.05 | 5.39 | 294.32 |
| Conv134 | 64 | 256 | 0.65 | 0.05 | 0.90 | 0.05 | 0.90 | 0.05 | 6.82 | 83.68 |
| Conv135 | 64 | 576 | 0.88 | 0.05 | 0.97 | 0.05 | 0.97 | 0.05 | 0.23 | 83.68 |
| Conv136 | 256 | 64 | 0.85 | 0.05 | 0.90 | 0.05 | 0.90 | 0.05 | 6.57 | 294.32 |
| Conv137 | 64 | 256 | 0.68 | 0.05 | 0.90 | 0.05 | 0.90 | 0.05 | 6.95 | 83.68 |
| Conv138 | 64 | 576 | 0.88 | 0.05 | 0.97 | 0.05 | 0.97 | 0.05 | 0.22 | 83.68 |
| Conv139 | 256 | 64 | 0.85 | 0.05 | 0.90 | 0.05 | 0.90 | 0.05 | 6.3 | 294.32 |
| Conv140 | 64 | 256 | 0.62 | 0.05 | 0.90 | 0.05 | 0.90 | 0.05 | 4.99 | 83.68 |
| Conv141 | 64 | 576 | 0.89 | 0.05 | 0.97 | 0.05 | 0.97 | 0.05 | 0.21 | 83.68 |
| Conv142 | 256 | 64 | 0.86 | 0.05 | 0.90 | 0.05 | 0.90 | 0.05 | 4.94 | 294.32 |
| Conv143 | 64 | 256 | 0.62 | 0.05 | 0.90 | 0.05 | 0.90 | 0.05 | 4.11 | 83.68 |
| Conv144 | 64 | 576 | 0.88 | 0.05 | 0.97 | 0.05 | 0.97 | 0.05 | 0.27 | 83.68 |
| Conv145 | 256 | 64 | 0.82 | 0.05 | 0.90 | 0.05 | 0.90 | 0.05 | 5.79 | 294.32 |
| Conv146 | 64 | 256 | 0.62 | 0.05 | 0.90 | 0.05 | 0.90 | 0.05 | 4.72 | 83.68 |
| Conv147 | 64 | 576 | 0.86 | 0.05 | 0.97 | 0.05 | 0.97 | 0.05 | 0.32 | 83.68 |
| Conv148 | 256 | 64 | 0.85 | 0.05 | 0.90 | 0.05 | 0.90 | 0.05 | 4.91 | 294.32 |
| Conv149 | 64 | 256 | 0.65 | 0.05 | 0.90 | 0.05 | 0.90 | 0.05 | 3.53 | 83.68 |
| Conv150 | 64 | 576 | 0.89 | 0.05 | 0.97 | 0.05 | 0.97 | 0.05 | 0.21 | 83.68 |
| Conv151 | 256 | 64 | 0.85 | 0.05 | 0.90 | 0.05 | 0.90 | 0.05 | 4.28 | 294.32 |
| Conv152 | 64 | 256 | 0.65 | 0.05 | 0.90 | 0.05 | 0.90 | 0.05 | 4.26 | 83.68 |
| Conv153 | 64 | 576 | 0.88 | 0.05 | 0.97 | 0.05 | 0.97 | 0.05 | 0.2 | 83.68 |
| Conv154 | 256 | 64 | 0.86 | 0.05 | 0.90 | 0.05 | 0.90 | 0.05 | 3.77 | 294.32 |
| Conv155 | 64 | 256 | 0.62 | 0.05 | 0.90 | 0.05 | 0.90 | 0.05 | 4.45 | 83.68 |
| Conv156 | 64 | 576 | 0.89 | 0.05 | 0.97 | 0.05 | 0.97 | 0.05 | 0.12 | 83.68 |
| Conv157 | 256 | 64 | 0.85 | 0.05 | 0.90 | 0.05 | 0.90 | 0.05 | 3.0 | 294.32 |
| Conv158 | 64 | 256 | 0.66 | 0.05 | 0.90 | 0.05 | 0.90 | 0.05 | 5.59 | 83.68 |
| Conv159 | 64 | 576 | 0.89 | 0.05 | 0.97 | 0.05 | 0.97 | 0.05 | 0.22 | 83.68 |
| Conv160 | 256 | 64 | 0.85 | 0.05 | 0.90 | 0.05 | 0.90 | 0.05 | 3.12 | 294.32 |
| Conv161 | 64 | 256 | 0.63 | 0.05 | 0.90 | 0.05 | 0.90 | 0.05 | 4.89 | 83.68 |
| Conv162 | 64 | 576 | 0.88 | 0.05 | 0.97 | 0.05 | 0.97 | 0.05 | 0.17 | 83.68 |
| Conv163 | 256 | 64 | 0.86 | 0.05 | 0.90 | 0.05 | 0.90 | 0.05 | 2.38 | 294.32 |
| **Passing rate** | - | - | 99.39% | | 100.0% | | 100.0% | | 99.39% | |

Table 18: Cifar100 Adamax-ResNet164

| Layer | Number | dim | Gaussian | | Mean_Left | | Mean_Right | | Sigma | |
|-------|--------|-----|---------|---------|-----------|---------|------------|---------|--------|---------|
| | | | p-value | c-value | p-value | c-value | p-value | c-value | t-value | c-value |
| Conv1 | 16 | 27 | 0.13 | 0.05 | 0.58 | 0.05 | 0.58 | 0.05 | 68.0 | 26.3 |
| Conv2 | 16 | 16 | 0.48 | 0.05 | 0.56 | 0.05 | 0.56 | 0.05 | 71.45 | 26.3 |
| Conv3 | 16 | 144 | 0.66 | 0.05 | 0.68 | 0.05 | 0.68 | 0.05 | 11.09 | 26.3 |
| Conv4 | 64 | 16 | 0.59 | 0.05 | 0.63 | 0.05 | 0.63 | 0.05 | 70.98 | 83.68 |
| Conv5 | 16 | 64 | 0.55 | 0.05 | 0.63 | 0.05 | 0.63 | 0.05 | 20.74 | 26.3 |
| Conv6 | 16 | 144 | 0.61 | 0.05 | 0.68 | 0.05 | 0.68 | 0.05 | 13.67 | 26.3 |

| | | | | | | | | | | |
|---|---|---|---|---|---|---|---|---|---|---|
| Conv7 | 64 | 16 | 0.63 | 0.05 | 0.63 | 0.05 | 0.63 | 0.05 | 78.55 | 83.68 |
| Conv8 | 16 | 64 | 0.46 | 0.05 | 0.63 | 0.05 | 0.63 | 0.05 | 13.55 | 26.3 |
| Conv9 | 16 | 144 | 0.69 | 0.05 | 0.68 | 0.05 | 0.68 | 0.05 | 7.71 | 26.3 |
| Conv10 | 64 | 16 | 0.70 | 0.05 | 0.63 | 0.05 | 0.63 | 0.05 | 21.48 | 83.68 |
| Conv11 | 16 | 64 | 0.66 | 0.05 | 0.63 | 0.05 | 0.63 | 0.05 | 4.16 | 26.3 |
| Conv12 | 16 | 144 | 0.89 | 0.05 | 0.68 | 0.05 | 0.68 | 0.05 | 1.72 | 26.3 |
| Conv13 | 64 | 16 | 0.88 | 0.05 | 0.63 | 0.05 | 0.63 | 0.05 | 7.36 | 83.68 |
| Conv14 | 16 | 64 | 0.69 | 0.05 | 0.63 | 0.05 | 0.63 | 0.05 | 8.77 | 26.3 |
| Conv15 | 16 | 144 | 0.67 | 0.05 | 0.68 | 0.05 | 0.68 | 0.05 | 2.77 | 26.3 |
| Conv16 | 64 | 16 | 0.80 | 0.05 | 0.63 | 0.05 | 0.63 | 0.05 | 7.36 | 83.68 |
| Conv17 | 16 | 64 | 0.70 | 0.05 | 0.63 | 0.05 | 0.63 | 0.05 | 6.74 | 26.3 |
| Conv18 | 16 | 144 | 0.71 | 0.05 | 0.68 | 0.05 | 0.68 | 0.05 | 2.63 | 26.3 |
| Conv19 | 64 | 16 | 0.82 | 0.05 | 0.63 | 0.05 | 0.63 | 0.05 | 11.07 | 83.68 |
| Conv20 | 16 | 64 | 0.44 | 0.05 | 0.63 | 0.05 | 0.63 | 0.05 | 15.46 | 26.3 |
| Conv21 | 16 | 144 | 0.51 | 0.05 | 0.68 | 0.05 | 0.68 | 0.05 | 7.75 | 26.3 |
| Conv22 | 64 | 16 | 0.66 | 0.05 | 0.63 | 0.05 | 0.63 | 0.05 | 24.59 | 83.68 |
| Conv23 | 16 | 64 | 0.54 | 0.05 | 0.63 | 0.05 | 0.63 | 0.05 | 16.27 | 26.3 |
| Conv24 | 16 | 144 | 0.62 | 0.05 | 0.68 | 0.05 | 0.68 | 0.05 | 10.25 | 26.3 |
| Conv25 | 64 | 16 | 0.71 | 0.05 | 0.63 | 0.05 | 0.63 | 0.05 | 44.15 | 83.68 |
| Conv26 | 16 | 64 | 0.68 | 0.05 | 0.63 | 0.05 | 0.63 | 0.05 | 12.42 | 26.3 |
| Conv27 | 16 | 144 | 0.64 | 0.05 | 0.68 | 0.05 | 0.68 | 0.05 | 4.52 | 26.3 |
| Conv28 | 64 | 16 | 0.80 | 0.05 | 0.63 | 0.05 | 0.63 | 0.05 | 21.59 | 83.68 |
| Conv29 | 16 | 64 | 0.54 | 0.05 | 0.63 | 0.05 | 0.63 | 0.05 | 19.57 | 26.3 |
| Conv30 | 16 | 144 | 0.60 | 0.05 | 0.68 | 0.05 | 0.68 | 0.05 | 10.01 | 26.3 |
| Conv31 | 64 | 16 | 0.63 | 0.05 | 0.63 | 0.05 | 0.63 | 0.05 | 50.96 | 83.68 |
| Conv32 | 16 | 64 | 0.41 | 0.05 | 0.63 | 0.05 | 0.63 | 0.05 | 15.87 | 26.3 |
| Conv33 | 16 | 144 | 0.58 | 0.05 | 0.68 | 0.05 | 0.68 | 0.05 | 6.75 | 26.3 |
| Conv34 | 64 | 16 | 0.66 | 0.05 | 0.63 | 0.05 | 0.63 | 0.05 | 52.95 | 83.68 |
| Conv35 | 16 | 64 | 0.59 | 0.05 | 0.63 | 0.05 | 0.63 | 0.05 | 10.96 | 26.3 |
| Conv36 | 16 | 144 | 0.66 | 0.05 | 0.68 | 0.05 | 0.68 | 0.05 | 4.91 | 26.3 |
| Conv37 | 64 | 16 | 0.65 | 0.05 | 0.63 | 0.05 | 0.63 | 0.05 | 21.71 | 83.68 |
| Conv38 | 16 | 64 | 0.35 | 0.05 | 0.63 | 0.05 | 0.63 | 0.05 | 24.32 | 26.3 |
| Conv39 | 16 | 144 | 0.50 | 0.05 | 0.68 | 0.05 | 0.68 | 0.05 | 8.62 | 26.3 |
| Conv40 | 64 | 16 | 0.67 | 0.05 | 0.63 | 0.05 | 0.63 | 0.05 | 49.57 | 83.68 |
| Conv41 | 16 | 64 | 0.61 | 0.05 | 0.63 | 0.05 | 0.63 | 0.05 | 16.45 | 26.3 |
| Conv42 | 16 | 144 | 0.63 | 0.05 | 0.68 | 0.05 | 0.68 | 0.05 | 9.32 | 26.3 |
| Conv43 | 64 | 16 | 0.68 | 0.05 | 0.63 | 0.05 | 0.63 | 0.05 | 38.26 | 83.68 |
| Conv44 | 16 | 64 | 0.68 | 0.05 | 0.63 | 0.05 | 0.63 | 0.05 | 12.91 | 26.3 |
| Conv45 | 16 | 144 | 0.65 | 0.05 | 0.68 | 0.05 | 0.68 | 0.05 | 8.54 | 26.3 |
| Conv46 | 64 | 16 | 0.65 | 0.05 | 0.63 | 0.05 | 0.63 | 0.05 | 53.09 | 83.68 |
| Conv47 | 16 | 64 | 0.55 | 0.05 | 0.63 | 0.05 | 0.63 | 0.05 | 16.67 | 26.3 |
| Conv48 | 16 | 144 | 0.56 | 0.05 | 0.68 | 0.05 | 0.68 | 0.05 | 8.19 | 26.3 |
| Conv49 | 64 | 16 | 0.66 | 0.05 | 0.63 | 0.05 | 0.63 | 0.05 | 39.79 | 83.68 |
| Conv50 | 16 | 64 | 0.59 | 0.05 | 0.63 | 0.05 | 0.63 | 0.05 | 17.17 | 26.3 |
| Conv51 | 16 | 144 | 0.61 | 0.05 | 0.68 | 0.05 | 0.68 | 0.05 | 10.77 | 26.3 |
| Conv52 | 64 | 16 | 0.61 | 0.05 | 0.63 | 0.05 | 0.63 | 0.05 | 51.64 | 83.68 |
| Conv53 | 16 | 64 | 0.37 | 0.05 | 0.63 | 0.05 | 0.63 | 0.05 | 24.49 | 26.3 |
| Conv54 | 16 | 144 | 0.57 | 0.05 | 0.68 | 0.05 | 0.68 | 0.05 | 7.39 | 26.3 |
| Conv55 | 64 | 16 | 0.62 | 0.05 | 0.63 | 0.05 | 0.63 | 0.05 | 45.71 | 83.68 |
| Conv56 | 32 | 64 | 0.35 | 0.05 | 0.67 | 0.05 | 0.67 | 0.05 | 84.41 | 46.19 |
| Conv57 | 32 | 288 | 0.56 | 0.05 | 0.83 | 0.05 | 0.83 | 0.05 | 19.39 | 46.19 |
| Conv58 | 128 | 32 | 0.42 | 0.05 | 0.74 | 0.05 | 0.74 | 0.05 | 198.37 | 155.4 |
| Conv59 | 32 | 128 | 0.43 | 0.05 | 0.74 | 0.05 | 0.74 | 0.05 | 16.2 | 46.19 |
| Conv60 | 32 | 288 | 0.39 | 0.05 | 0.83 | 0.05 | 0.83 | 0.05 | 10.01 | 46.19 |
| Conv61 | 128 | 32 | 0.62 | 0.05 | 0.74 | 0.05 | 0.74 | 0.05 | 49.07 | 155.4 |
| Conv62 | 32 | 128 | 0.43 | 0.05 | 0.74 | 0.05 | 0.74 | 0.05 | 11.75 | 46.19 |
| Conv63 | 32 | 288 | 0.55 | 0.05 | 0.83 | 0.05 | 0.83 | 0.05 | 6.82 | 46.19 |
| Conv64 | 128 | 32 | 0.62 | 0.05 | 0.74 | 0.05 | 0.74 | 0.05 | 37.0 | 155.4 |
| Conv65 | 32 | 128 | 0.54 | 0.05 | 0.74 | 0.05 | 0.74 | 0.05 | 8.23 | 46.19 |

| | | | | | | | | | | |
|---|---|---|---|---|---|---|---|---|---|---|
| Conv66 | 32 | 288 | 0.45 | 0.05 | 0.83 | 0.05 | 0.83 | 0.05 | 5.49 | 46.19 |
| Conv67 | 128 | 32 | 0.69 | 0.05 | 0.74 | 0.05 | 0.74 | 0.05 | 34.13 | 155.4 |
| Conv68 | 32 | 128 | 0.41 | 0.05 | 0.74 | 0.05 | 0.74 | 0.05 | 9.57 | 46.19 |
| Conv69 | 32 | 288 | 0.49 | 0.05 | 0.83 | 0.05 | 0.83 | 0.05 | 6.21 | 46.19 |
| Conv70 | 128 | 32 | 0.69 | 0.05 | 0.74 | 0.05 | 0.74 | 0.05 | 28.63 | 155.4 |
| Conv71 | 32 | 128 | 0.38 | 0.05 | 0.74 | 0.05 | 0.74 | 0.05 | 12.89 | 46.19 |
| Conv72 | 32 | 288 | 0.42 | 0.05 | 0.83 | 0.05 | 0.83 | 0.05 | 7.03 | 46.19 |
| Conv73 | 128 | 32 | 0.59 | 0.05 | 0.74 | 0.05 | 0.74 | 0.05 | 34.59 | 155.4 |
| Conv74 | 32 | 128 | 0.50 | 0.05 | 0.74 | 0.05 | 0.74 | 0.05 | 18.97 | 46.19 |
| Conv75 | 32 | 288 | 0.68 | 0.05 | 0.83 | 0.05 | 0.83 | 0.05 | 8.88 | 46.19 |
| Conv76 | 128 | 32 | 0.69 | 0.05 | 0.74 | 0.05 | 0.74 | 0.05 | 40.73 | 155.4 |
| Conv77 | 32 | 128 | 0.57 | 0.05 | 0.74 | 0.05 | 0.74 | 0.05 | 12.85 | 46.19 |
| Conv78 | 32 | 288 | 0.64 | 0.05 | 0.83 | 0.05 | 0.83 | 0.05 | 4.65 | 46.19 |
| Conv79 | 128 | 32 | 0.82 | 0.05 | 0.74 | 0.05 | 0.74 | 0.05 | 23.62 | 155.4 |
| Conv80 | 32 | 128 | 0.59 | 0.05 | 0.74 | 0.05 | 0.74 | 0.05 | 8.78 | 46.19 |
| Conv81 | 32 | 288 | 0.61 | 0.05 | 0.83 | 0.05 | 0.83 | 0.05 | 3.45 | 46.19 |
| Conv82 | 128 | 32 | 0.81 | 0.05 | 0.74 | 0.05 | 0.74 | 0.05 | 17.55 | 155.4 |
| Conv83 | 32 | 128 | 0.55 | 0.05 | 0.74 | 0.05 | 0.74 | 0.05 | 16.4 | 46.19 |
| Conv84 | 32 | 288 | 0.56 | 0.05 | 0.83 | 0.05 | 0.83 | 0.05 | 8.33 | 46.19 |
| Conv85 | 128 | 32 | 0.66 | 0.05 | 0.74 | 0.05 | 0.74 | 0.05 | 38.5 | 155.4 |
| Conv86 | 32 | 128 | 0.39 | 0.05 | 0.74 | 0.05 | 0.74 | 0.05 | 23.91 | 46.19 |
| Conv87 | 32 | 288 | 0.56 | 0.05 | 0.83 | 0.05 | 0.83 | 0.05 | 12.35 | 46.19 |
| Conv88 | 128 | 32 | 0.54 | 0.05 | 0.74 | 0.05 | 0.74 | 0.05 | 85.92 | 155.4 |
| Conv89 | 32 | 128 | 0.56 | 0.05 | 0.74 | 0.05 | 0.74 | 0.05 | 20.14 | 46.19 |
| Conv90 | 32 | 288 | 0.63 | 0.05 | 0.83 | 0.05 | 0.83 | 0.05 | 8.2 | 46.19 |
| Conv91 | 128 | 32 | 0.69 | 0.05 | 0.74 | 0.05 | 0.74 | 0.05 | 47.23 | 155.4 |
| Conv92 | 32 | 128 | 0.54 | 0.05 | 0.74 | 0.05 | 0.74 | 0.05 | 11.75 | 46.19 |
| Conv93 | 32 | 288 | 0.58 | 0.05 | 0.83 | 0.05 | 0.83 | 0.05 | 5.63 | 46.19 |
| Conv94 | 128 | 32 | 0.81 | 0.05 | 0.74 | 0.05 | 0.74 | 0.05 | 25.93 | 155.4 |
| Conv95 | 32 | 128 | 0.61 | 0.05 | 0.74 | 0.05 | 0.74 | 0.05 | 16.71 | 46.19 |
| Conv96 | 32 | 288 | 0.68 | 0.05 | 0.83 | 0.05 | 0.83 | 0.05 | 7.92 | 46.19 |
| Conv97 | 128 | 32 | 0.69 | 0.05 | 0.74 | 0.05 | 0.74 | 0.05 | 37.55 | 155.4 |
| Conv98 | 32 | 128 | 0.40 | 0.05 | 0.74 | 0.05 | 0.74 | 0.05 | 22.82 | 46.19 |
| Conv99 | 32 | 288 | 0.80 | 0.05 | 0.83 | 0.05 | 0.83 | 0.05 | 6.79 | 46.19 |
| Conv100 | 128 | 32 | 0.67 | 0.05 | 0.74 | 0.05 | 0.74 | 0.05 | 49.54 | 155.4 |
| Conv101 | 32 | 128 | 0.62 | 0.05 | 0.74 | 0.05 | 0.74 | 0.05 | 19.77 | 46.19 |
| Conv102 | 32 | 288 | 0.67 | 0.05 | 0.83 | 0.05 | 0.83 | 0.05 | 7.45 | 46.19 |
| Conv103 | 128 | 32 | 0.62 | 0.05 | 0.74 | 0.05 | 0.74 | 0.05 | 56.52 | 155.4 |
| Conv104 | 32 | 128 | 0.47 | 0.05 | 0.74 | 0.05 | 0.74 | 0.05 | 19.67 | 46.19 |
| Conv105 | 32 | 288 | 0.59 | 0.05 | 0.83 | 0.05 | 0.83 | 0.05 | 11.28 | 46.19 |
| Conv106 | 128 | 32 | 0.66 | 0.05 | 0.74 | 0.05 | 0.74 | 0.05 | 55.28 | 155.4 |
| Conv107 | 32 | 128 | 0.53 | 0.05 | 0.74 | 0.05 | 0.74 | 0.05 | 20.02 | 46.19 |
| Conv108 | 32 | 288 | 0.63 | 0.05 | 0.83 | 0.05 | 0.83 | 0.05 | 6.73 | 46.19 |
| Conv109 | 128 | 32 | 0.67 | 0.05 | 0.74 | 0.05 | 0.74 | 0.05 | 38.49 | 155.4 |
| Conv110 | 64 | 128 | 0.30 | 0.05 | 0.82 | 0.05 | 0.82 | 0.05 | 102.5 | 83.68 |
| Conv111 | 64 | 576 | 0.51 | 0.05 | 0.97 | 0.05 | 0.97 | 0.05 | 20.27 | 83.68 |
| Conv112 | 256 | 64 | 0.18 | 0.05 | 0.90 | 0.05 | 0.90 | 0.05 | 228.72 | 294.32 |
| Conv113 | 64 | 256 | 0.63 | 0.05 | 0.90 | 0.05 | 0.90 | 0.05 | 19.19 | 83.68 |
| Conv114 | 64 | 576 | 0.51 | 0.05 | 0.97 | 0.05 | 0.97 | 0.05 | 13.0 | 83.68 |
| Conv115 | 256 | 64 | 0.59 | 0.05 | 0.90 | 0.05 | 0.90 | 0.05 | 59.58 | 294.32 |
| Conv116 | 64 | 256 | 0.67 | 0.05 | 0.90 | 0.05 | 0.90 | 0.05 | 18.46 | 83.68 |
| Conv117 | 64 | 576 | 0.47 | 0.05 | 0.97 | 0.05 | 0.97 | 0.05 | 11.64 | 83.68 |
| Conv118 | 256 | 64 | 0.64 | 0.05 | 0.90 | 0.05 | 0.90 | 0.05 | 59.16 | 294.32 |
| Conv119 | 64 | 256 | 0.65 | 0.05 | 0.90 | 0.05 | 0.90 | 0.05 | 18.95 | 83.68 |
| Conv120 | 64 | 576 | 0.44 | 0.05 | 0.97 | 0.05 | 0.97 | 0.05 | 12.67 | 83.68 |
| Conv121 | 256 | 64 | 0.63 | 0.05 | 0.90 | 0.05 | 0.90 | 0.05 | 47.73 | 294.32 |
| Conv122 | 64 | 256 | 0.54 | 0.05 | 0.90 | 0.05 | 0.90 | 0.05 | 19.16 | 83.68 |
| Conv123 | 64 | 576 | 0.63 | 0.05 | 0.97 | 0.05 | 0.97 | 0.05 | 10.26 | 83.68 |
| Conv124 | 256 | 64 | 0.81 | 0.05 | 0.90 | 0.05 | 0.90 | 0.05 | 42.2 | 294.32 |

| Layer | Number | dim | Gaussian | | Mean_Left | | Mean_Right | | Sigma | |
|---|---|---|---|---|---|---|---|---|---|---|
| | | | p-value | c-value | p-value | c-value | p-value | c-value | t-value | c-value |
| Conv125 | 64 | 256 | 0.59 | 0.05 | 0.90 | 0.05 | 0.90 | 0.05 | 24.01 | 83.68 |
| Conv126 | 64 | 576 | 0.69 | 0.05 | 0.97 | 0.05 | 0.97 | 0.05 | 11.76 | 83.68 |
| Conv127 | 256 | 64 | 0.65 | 0.05 | 0.90 | 0.05 | 0.90 | 0.05 | 39.33 | 294.32 |
| Conv128 | 64 | 256 | 0.28 | 0.05 | 0.90 | 0.05 | 0.90 | 0.05 | 25.97 | 83.68 |
| Conv129 | 64 | 576 | 0.60 | 0.05 | 0.97 | 0.05 | 0.97 | 0.05 | 13.68 | 83.68 |
| Conv130 | 256 | 64 | 0.51 | 0.05 | 0.90 | 0.05 | 0.90 | 0.05 | 62.74 | 294.32 |
| Conv131 | 64 | 256 | 0.18 | 0.05 | 0.90 | 0.05 | 0.90 | 0.05 | 37.84 | 83.68 |
| Conv132 | 64 | 576 | 0.55 | 0.05 | 0.97 | 0.05 | 0.97 | 0.05 | 16.28 | 83.68 |
| Conv133 | 256 | 64 | 0.43 | 0.05 | 0.90 | 0.05 | 0.90 | 0.05 | 98.19 | 294.32 |
| Conv134 | 64 | 256 | 0.43 | 0.05 | 0.90 | 0.05 | 0.90 | 0.05 | 39.47 | 83.68 |
| Conv135 | 64 | 576 | 0.47 | 0.05 | 0.97 | 0.05 | 0.97 | 0.05 | 13.86 | 83.68 |
| Conv136 | 256 | 64 | 0.71 | 0.05 | 0.90 | 0.05 | 0.90 | 0.05 | 65.95 | 294.32 |
| Conv137 | 64 | 256 | 0.64 | 0.05 | 0.90 | 0.05 | 0.90 | 0.05 | 44.63 | 83.68 |
| Conv138 | 64 | 576 | 0.65 | 0.05 | 0.97 | 0.05 | 0.97 | 0.05 | 13.91 | 83.68 |
| Conv139 | 256 | 64 | 0.53 | 0.05 | 0.90 | 0.05 | 0.90 | 0.05 | 70.45 | 294.32 |
| Conv140 | 64 | 256 | 0.59 | 0.05 | 0.90 | 0.05 | 0.90 | 0.05 | 44.73 | 83.68 |
| Conv141 | 64 | 576 | 0.68 | 0.05 | 0.97 | 0.05 | 0.97 | 0.05 | 11.5 | 83.68 |
| Conv142 | 256 | 64 | 0.55 | 0.05 | 0.90 | 0.05 | 0.90 | 0.05 | 84.15 | 294.32 |
| Conv143 | 64 | 256 | 0.52 | 0.05 | 0.90 | 0.05 | 0.90 | 0.05 | 45.81 | 83.68 |
| Conv144 | 64 | 576 | 0.64 | 0.05 | 0.97 | 0.05 | 0.97 | 0.05 | 2.99 | 83.68 |
| Conv145 | 256 | 64 | 0.58 | 0.05 | 0.90 | 0.05 | 0.90 | 0.05 | 80.95 | 294.32 |
| Conv146 | 64 | 256 | 0.49 | 0.05 | 0.90 | 0.05 | 0.90 | 0.05 | 43.23 | 83.68 |
| Conv147 | 64 | 576 | 0.67 | 0.05 | 0.97 | 0.05 | 0.97 | 0.05 | 11.54 | 83.68 |
| Conv148 | 256 | 64 | 0.60 | 0.05 | 0.90 | 0.05 | 0.90 | 0.05 | 76.94 | 294.32 |
| Conv149 | 64 | 256 | 0.53 | 0.05 | 0.90 | 0.05 | 0.90 | 0.05 | 30.26 | 83.68 |
| Conv150 | 64 | 576 | 0.69 | 0.05 | 0.97 | 0.05 | 0.97 | 0.05 | 6.73 | 83.68 |
| Conv151 | 256 | 64 | 0.48 | 0.05 | 0.90 | 0.05 | 0.90 | 0.05 | 81.75 | 294.32 |
| Conv152 | 64 | 256 | 0.56 | 0.05 | 0.90 | 0.05 | 0.90 | 0.05 | 33.02 | 83.68 |
| Conv153 | 64 | 576 | 0.81 | 0.05 | 0.97 | 0.05 | 0.97 | 0.05 | 8.73 | 83.68 |
| Conv154 | 256 | 64 | 0.55 | 0.05 | 0.90 | 0.05 | 0.90 | 0.05 | 91.86 | 294.32 |
| Conv155 | 64 | 256 | 0.59 | 0.05 | 0.90 | 0.05 | 0.90 | 0.05 | 29.47 | 83.68 |
| Conv156 | 64 | 576 | 0.83 | 0.05 | 0.97 | 0.05 | 0.97 | 0.05 | 11.58 | 83.68 |
| Conv157 | 256 | 64 | 0.60 | 0.05 | 0.90 | 0.05 | 0.90 | 0.05 | 76.05 | 294.32 |
| Conv158 | 64 | 256 | 0.47 | 0.05 | 0.90 | 0.05 | 0.90 | 0.05 | 41.88 | 83.68 |
| Conv159 | 64 | 576 | 0.82 | 0.05 | 0.97 | 0.05 | 0.97 | 0.05 | 15.7 | 83.68 |
| Conv160 | 256 | 64 | 0.53 | 0.05 | 0.90 | 0.05 | 0.90 | 0.05 | 64.48 | 294.32 |
| Conv161 | 64 | 256 | 0.51 | 0.05 | 0.90 | 0.05 | 0.90 | 0.05 | 68.02 | 83.68 |
| Conv162 | 64 | 576 | 0.81 | 0.05 | 0.97 | 0.05 | 0.97 | 0.05 | 12.04 | 83.68 |
| Conv163 | 256 | 64 | 0.61 | 0.05 | 0.90 | 0.05 | 0.90 | 0.05 | 51.41 | 294.32 |
| **Passing rate** | - | - | 100.0% | | 100.0% | | 100.0% | | 96.93% | |

Table 19: Cifar100 Adadelta-ResNet164

| Layer | Number | dim | Gaussian | | Mean_Left | | Mean_Right | | Sigma | |
|---|---|---|---|---|---|---|---|---|---|---|
| | | | p-value | c-value | p-value | c-value | p-value | c-value | t-value | c-value |
| Conv1 | 16 | 27 | 0.11 | 0.05 | 0.58 | 0.05 | 0.58 | 0.05 | 8.49 | 26.3 |
| Conv2 | 16 | 16 | 0.43 | 0.05 | 0.56 | 0.05 | 0.56 | 0.05 | 9.33 | 26.3 |
| Conv3 | 16 | 144 | 0.56 | 0.05 | 0.68 | 0.05 | 0.68 | 0.05 | 4.11 | 26.3 |
| Conv4 | 64 | 16 | 0.44 | 0.05 | 0.63 | 0.05 | 0.63 | 0.05 | 45.93 | 83.68 |
| Conv5 | 16 | 64 | 0.53 | 0.05 | 0.63 | 0.05 | 0.63 | 0.05 | 3.17 | 26.3 |
| Conv6 | 16 | 144 | 0.56 | 0.05 | 0.68 | 0.05 | 0.68 | 0.05 | 6.72 | 26.3 |
| Conv7 | 64 | 16 | 0.64 | 0.05 | 0.63 | 0.05 | 0.63 | 0.05 | 23.93 | 83.68 |
| Conv8 | 16 | 64 | 0.62 | 0.05 | 0.63 | 0.05 | 0.63 | 0.05 | 3.05 | 26.3 |
| Conv9 | 16 | 144 | 0.67 | 0.05 | 0.68 | 0.05 | 0.68 | 0.05 | 2.49 | 26.3 |
| Conv10 | 64 | 16 | 0.67 | 0.05 | 0.63 | 0.05 | 0.63 | 0.05 | 24.31 | 83.68 |

| Conv11 | 16 | 64 | 0.44 | 0.05 | 0.63 | 0.05 | 0.63 | 0.05 | 8.09 | 26.3 |
|---|---|---|---|---|---|---|---|---|---|---|
| Conv12 | 16 | 144 | 0.43 | 0.05 | 0.68 | 0.05 | 0.68 | 0.05 | 4.44 | 26.3 |
| Conv13 | 64 | 16 | 0.59 | 0.05 | 0.63 | 0.05 | 0.63 | 0.05 | 26.43 | 83.68 |
| Conv14 | 16 | 64 | 0.61 | 0.05 | 0.63 | 0.05 | 0.63 | 0.05 | 3.06 | 26.3 |
| Conv15 | 16 | 144 | 0.54 | 0.05 | 0.68 | 0.05 | 0.68 | 0.05 | 2.9 | 26.3 |
| Conv16 | 64 | 16 | 0.62 | 0.05 | 0.63 | 0.05 | 0.63 | 0.05 | 21.66 | 83.68 |
| Conv17 | 16 | 64 | 0.63 | 0.05 | 0.63 | 0.05 | 0.63 | 0.05 | 2.38 | 26.3 |
| Conv18 | 16 | 144 | 0.66 | 0.05 | 0.68 | 0.05 | 0.68 | 0.05 | 2.37 | 26.3 |
| Conv19 | 64 | 16 | 0.62 | 0.05 | 0.63 | 0.05 | 0.63 | 0.05 | 22.61 | 83.68 |
| Conv20 | 16 | 64 | 0.65 | 0.05 | 0.63 | 0.05 | 0.63 | 0.05 | 2.13 | 26.3 |
| Conv21 | 16 | 144 | 0.81 | 0.05 | 0.68 | 0.05 | 0.68 | 0.05 | 1.53 | 26.3 |
| Conv22 | 64 | 16 | 0.80 | 0.05 | 0.63 | 0.05 | 0.63 | 0.05 | 17.96 | 83.68 |
| Conv23 | 16 | 64 | 0.62 | 0.05 | 0.63 | 0.05 | 0.63 | 0.05 | 3.82 | 26.3 |
| Conv24 | 16 | 144 | 0.64 | 0.05 | 0.68 | 0.05 | 0.68 | 0.05 | 1.5 | 26.3 |
| Conv25 | 64 | 16 | 0.70 | 0.05 | 0.63 | 0.05 | 0.63 | 0.05 | 13.74 | 83.68 |
| Conv26 | 16 | 64 | 0.59 | 0.05 | 0.63 | 0.05 | 0.63 | 0.05 | 3.75 | 26.3 |
| Conv27 | 16 | 144 | 0.62 | 0.05 | 0.68 | 0.05 | 0.68 | 0.05 | 1.67 | 26.3 |
| Conv28 | 64 | 16 | 0.63 | 0.05 | 0.63 | 0.05 | 0.63 | 0.05 | 14.36 | 83.68 |
| Conv29 | 16 | 64 | 0.67 | 0.05 | 0.63 | 0.05 | 0.63 | 0.05 | 1.62 | 26.3 |
| Conv30 | 16 | 144 | 0.68 | 0.05 | 0.68 | 0.05 | 0.68 | 0.05 | 1.75 | 26.3 |
| Conv31 | 64 | 16 | 0.80 | 0.05 | 0.63 | 0.05 | 0.63 | 0.05 | 11.87 | 83.68 |
| Conv32 | 16 | 64 | 0.61 | 0.05 | 0.63 | 0.05 | 0.63 | 0.05 | 3.11 | 26.3 |
| Conv33 | 16 | 144 | 0.69 | 0.05 | 0.68 | 0.05 | 0.68 | 0.05 | 1.36 | 26.3 |
| Conv34 | 64 | 16 | 0.71 | 0.05 | 0.63 | 0.05 | 0.63 | 0.05 | 13.17 | 83.68 |
| Conv35 | 16 | 64 | 0.80 | 0.05 | 0.63 | 0.05 | 0.63 | 0.05 | 1.92 | 26.3 |
| Conv36 | 16 | 144 | 0.70 | 0.05 | 0.68 | 0.05 | 0.68 | 0.05 | 2.08 | 26.3 |
| Conv37 | 64 | 16 | 0.70 | 0.05 | 0.63 | 0.05 | 0.63 | 0.05 | 12.17 | 83.68 |
| Conv38 | 16 | 64 | 0.57 | 0.05 | 0.63 | 0.05 | 0.63 | 0.05 | 5.96 | 26.3 |
| Conv39 | 16 | 144 | 0.67 | 0.05 | 0.68 | 0.05 | 0.68 | 0.05 | 1.45 | 26.3 |
| Conv40 | 64 | 16 | 0.80 | 0.05 | 0.63 | 0.05 | 0.63 | 0.05 | 11.82 | 83.68 |
| Conv41 | 16 | 64 | 0.46 | 0.05 | 0.63 | 0.05 | 0.63 | 0.05 | 5.85 | 26.3 |
| Conv42 | 16 | 144 | 0.81 | 0.05 | 0.68 | 0.05 | 0.68 | 0.05 | 1.76 | 26.3 |
| Conv43 | 64 | 16 | 0.67 | 0.05 | 0.63 | 0.05 | 0.63 | 0.05 | 13.14 | 83.68 |
| Conv44 | 16 | 64 | 0.81 | 0.05 | 0.63 | 0.05 | 0.63 | 0.05 | 2.09 | 26.3 |
| Conv45 | 16 | 144 | 0.82 | 0.05 | 0.68 | 0.05 | 0.68 | 0.05 | 1.25 | 26.3 |
| Conv46 | 64 | 16 | 0.67 | 0.05 | 0.63 | 0.05 | 0.63 | 0.05 | 9.63 | 83.68 |
| Conv47 | 16 | 64 | 0.70 | 0.05 | 0.63 | 0.05 | 0.63 | 0.05 | 3.38 | 26.3 |
| Conv48 | 16 | 144 | 0.68 | 0.05 | 0.68 | 0.05 | 0.68 | 0.05 | 1.92 | 26.3 |
| Conv49 | 64 | 16 | 0.81 | 0.05 | 0.63 | 0.05 | 0.63 | 0.05 | 12.43 | 83.68 |
| Conv50 | 16 | 64 | 0.52 | 0.05 | 0.63 | 0.05 | 0.63 | 0.05 | 5.57 | 26.3 |
| Conv51 | 16 | 144 | 0.62 | 0.05 | 0.68 | 0.05 | 0.68 | 0.05 | 2.08 | 26.3 |
| Conv52 | 64 | 16 | 0.70 | 0.05 | 0.63 | 0.05 | 0.63 | 0.05 | 10.8 | 83.68 |
| Conv53 | 16 | 64 | 0.62 | 0.05 | 0.63 | 0.05 | 0.63 | 0.05 | 4.59 | 26.3 |
| Conv54 | 16 | 144 | 0.62 | 0.05 | 0.68 | 0.05 | 0.68 | 0.05 | 2.06 | 26.3 |
| Conv55 | 64 | 16 | 0.80 | 0.05 | 0.63 | 0.05 | 0.63 | 0.05 | 10.21 | 83.68 |
| Conv56 | 32 | 64 | 0.45 | 0.05 | 0.67 | 0.05 | 0.67 | 0.05 | 10.67 | 46.19 |
| Conv57 | 32 | 288 | 0.65 | 0.05 | 0.83 | 0.05 | 0.83 | 0.05 | 1.79 | 46.19 |
| Conv58 | 128 | 32 | 0.41 | 0.05 | 0.74 | 0.05 | 0.74 | 0.05 | 56.17 | 155.4 |
| Conv59 | 32 | 128 | 0.81 | 0.05 | 0.74 | 0.05 | 0.74 | 0.05 | 2.03 | 46.19 |
| Conv60 | 32 | 288 | 0.70 | 0.05 | 0.83 | 0.05 | 0.83 | 0.05 | 1.72 | 46.19 |
| Conv61 | 128 | 32 | 0.62 | 0.05 | 0.74 | 0.05 | 0.74 | 0.05 | 19.79 | 155.4 |
| Conv62 | 32 | 128 | 0.83 | 0.05 | 0.74 | 0.05 | 0.74 | 0.05 | 1.82 | 46.19 |
| Conv63 | 32 | 288 | 0.80 | 0.05 | 0.83 | 0.05 | 0.83 | 0.05 | 1.2 | 46.19 |
| Conv64 | 128 | 32 | 0.64 | 0.05 | 0.74 | 0.05 | 0.74 | 0.05 | 16.34 | 155.4 |
| Conv65 | 32 | 128 | 0.58 | 0.05 | 0.74 | 0.05 | 0.74 | 0.05 | 2.33 | 46.19 |
| Conv66 | 32 | 288 | 0.71 | 0.05 | 0.83 | 0.05 | 0.83 | 0.05 | 1.5 | 46.19 |
| Conv67 | 128 | 32 | 0.61 | 0.05 | 0.74 | 0.05 | 0.74 | 0.05 | 12.64 | 155.4 |
| Conv68 | 32 | 128 | 0.58 | 0.05 | 0.74 | 0.05 | 0.74 | 0.05 | 2.18 | 46.19 |
| Conv69 | 32 | 288 | 0.70 | 0.05 | 0.83 | 0.05 | 0.83 | 0.05 | 2.15 | 46.19 |

| Conv70 | 128 | 32 | 0.71 | 0.05 | 0.74 | 0.05 | 0.74 | 0.05 | 12.57 | 155.4 |
|---|---|---|---|---|---|---|---|---|---|---|
| Conv71 | 32 | 128 | 0.68 | 0.05 | 0.74 | 0.05 | 0.74 | 0.05 | 2.74 | 46.19 |
| Conv72 | 32 | 288 | 0.66 | 0.05 | 0.83 | 0.05 | 0.83 | 0.05 | 1.38 | 46.19 |
| Conv73 | 128 | 32 | 0.67 | 0.05 | 0.74 | 0.05 | 0.74 | 0.05 | 12.65 | 155.4 |
| Conv74 | 32 | 128 | 0.53 | 0.05 | 0.74 | 0.05 | 0.74 | 0.05 | 4.58 | 46.19 |
| Conv75 | 32 | 288 | 0.68 | 0.05 | 0.83 | 0.05 | 0.83 | 0.05 | 1.19 | 46.19 |
| Conv76 | 128 | 32 | 0.61 | 0.05 | 0.74 | 0.05 | 0.74 | 0.05 | 17.44 | 155.4 |
| Conv77 | 32 | 128 | 0.67 | 0.05 | 0.74 | 0.05 | 0.74 | 0.05 | 1.33 | 46.19 |
| Conv78 | 32 | 288 | 0.70 | 0.05 | 0.83 | 0.05 | 0.83 | 0.05 | 0.77 | 46.19 |
| Conv79 | 128 | 32 | 0.68 | 0.05 | 0.74 | 0.05 | 0.74 | 0.05 | 14.49 | 155.4 |
| Conv80 | 32 | 128 | 0.69 | 0.05 | 0.74 | 0.05 | 0.74 | 0.05 | 2.1 | 46.19 |
| Conv81 | 32 | 288 | 0.83 | 0.05 | 0.83 | 0.05 | 0.83 | 0.05 | 1.19 | 46.19 |
| Conv82 | 128 | 32 | 0.63 | 0.05 | 0.74 | 0.05 | 0.74 | 0.05 | 8.48 | 155.4 |
| Conv83 | 32 | 128 | 0.83 | 0.05 | 0.74 | 0.05 | 0.74 | 0.05 | 1.89 | 46.19 |
| Conv84 | 32 | 288 | 0.83 | 0.05 | 0.83 | 0.05 | 0.83 | 0.05 | 1.3 | 46.19 |
| Conv85 | 128 | 32 | 0.81 | 0.05 | 0.74 | 0.05 | 0.74 | 0.05 | 10.26 | 155.4 |
| Conv86 | 32 | 128 | 0.70 | 0.05 | 0.74 | 0.05 | 0.74 | 0.05 | 2.94 | 46.19 |
| Conv87 | 32 | 288 | 0.83 | 0.05 | 0.83 | 0.05 | 0.83 | 0.05 | 0.79 | 46.19 |
| Conv88 | 128 | 32 | 0.83 | 0.05 | 0.74 | 0.05 | 0.74 | 0.05 | 11.34 | 155.4 |
| Conv89 | 32 | 128 | 0.82 | 0.05 | 0.74 | 0.05 | 0.74 | 0.05 | 3.05 | 46.19 |
| Conv90 | 32 | 288 | 0.83 | 0.05 | 0.83 | 0.05 | 0.83 | 0.05 | 1.1 | 46.19 |
| Conv91 | 128 | 32 | 0.84 | 0.05 | 0.74 | 0.05 | 0.74 | 0.05 | 10.63 | 155.4 |
| Conv92 | 32 | 128 | 0.70 | 0.05 | 0.74 | 0.05 | 0.74 | 0.05 | 1.86 | 46.19 |
| Conv93 | 32 | 288 | 0.81 | 0.05 | 0.83 | 0.05 | 0.83 | 0.05 | 0.7 | 46.19 |
| Conv94 | 128 | 32 | 0.84 | 0.05 | 0.74 | 0.05 | 0.74 | 0.05 | 11.17 | 155.4 |
| Conv95 | 32 | 128 | 0.81 | 0.05 | 0.74 | 0.05 | 0.74 | 0.05 | 2.22 | 46.19 |
| Conv96 | 32 | 288 | 0.85 | 0.05 | 0.83 | 0.05 | 0.83 | 0.05 | 0.9 | 46.19 |
| Conv97 | 128 | 32 | 0.69 | 0.05 | 0.74 | 0.05 | 0.74 | 0.05 | 10.03 | 155.4 |
| Conv98 | 32 | 128 | 0.68 | 0.05 | 0.74 | 0.05 | 0.74 | 0.05 | 2.21 | 46.19 |
| Conv99 | 32 | 288 | 0.85 | 0.05 | 0.83 | 0.05 | 0.83 | 0.05 | 0.6 | 46.19 |
| Conv100 | 128 | 32 | 0.70 | 0.05 | 0.74 | 0.05 | 0.74 | 0.05 | 12.43 | 155.4 |
| Conv101 | 32 | 128 | 0.68 | 0.05 | 0.74 | 0.05 | 0.74 | 0.05 | 4.78 | 46.19 |
| Conv102 | 32 | 288 | 0.68 | 0.05 | 0.83 | 0.05 | 0.83 | 0.05 | 0.57 | 46.19 |
| Conv103 | 128 | 32 | 0.81 | 0.05 | 0.74 | 0.05 | 0.74 | 0.05 | 11.88 | 155.4 |
| Conv104 | 32 | 128 | 0.61 | 0.05 | 0.74 | 0.05 | 0.74 | 0.05 | 1.83 | 46.19 |
| Conv105 | 32 | 288 | 0.87 | 0.05 | 0.83 | 0.05 | 0.83 | 0.05 | 1.61 | 46.19 |
| Conv106 | 128 | 32 | 0.80 | 0.05 | 0.74 | 0.05 | 0.74 | 0.05 | 12.72 | 155.4 |
| Conv107 | 32 | 128 | 0.60 | 0.05 | 0.74 | 0.05 | 0.74 | 0.05 | 1.77 | 46.19 |
| Conv108 | 32 | 288 | 0.82 | 0.05 | 0.83 | 0.05 | 0.83 | 0.05 | 0.81 | 46.19 |
| Conv109 | 128 | 32 | 0.82 | 0.05 | 0.74 | 0.05 | 0.74 | 0.05 | 11.31 | 155.4 |
| Conv110 | 64 | 128 | 0.50 | 0.05 | 0.82 | 0.05 | 0.82 | 0.05 | 5.15 | 83.68 |
| Conv111 | 64 | 576 | 0.81 | 0.05 | 0.97 | 0.05 | 0.97 | 0.05 | 1.7 | 83.68 |
| Conv112 | 256 | 64 | 0.51 | 0.05 | 0.90 | 0.05 | 0.90 | 0.05 | 50.78 | 294.32 |
| Conv113 | 64 | 256 | 0.64 | 0.05 | 0.90 | 0.05 | 0.90 | 0.05 | 3.01 | 83.68 |
| Conv114 | 64 | 576 | 0.59 | 0.05 | 0.97 | 0.05 | 0.97 | 0.05 | 2.53 | 83.68 |
| Conv115 | 256 | 64 | 0.60 | 0.05 | 0.90 | 0.05 | 0.90 | 0.05 | 26.61 | 294.32 |
| Conv116 | 64 | 256 | 0.47 | 0.05 | 0.90 | 0.05 | 0.90 | 0.05 | 3.4 | 83.68 |
| Conv117 | 64 | 576 | 0.67 | 0.05 | 0.97 | 0.05 | 0.97 | 0.05 | 1.73 | 83.68 |
| Conv118 | 256 | 64 | 0.66 | 0.05 | 0.90 | 0.05 | 0.90 | 0.05 | 21.71 | 294.32 |
| Conv119 | 64 | 256 | 0.81 | 0.05 | 0.90 | 0.05 | 0.90 | 0.05 | 2.6 | 83.68 |
| Conv120 | 64 | 576 | 0.69 | 0.05 | 0.97 | 0.05 | 0.97 | 0.05 | 1.49 | 83.68 |
| Conv121 | 256 | 64 | 0.57 | 0.05 | 0.90 | 0.05 | 0.90 | 0.05 | 21.92 | 294.32 |
| Conv122 | 64 | 256 | 0.67 | 0.05 | 0.90 | 0.05 | 0.90 | 0.05 | 2.51 | 83.68 |
| Conv123 | 64 | 576 | 0.81 | 0.05 | 0.97 | 0.05 | 0.97 | 0.05 | 0.8 | 83.68 |
| Conv124 | 256 | 64 | 0.70 | 0.05 | 0.90 | 0.05 | 0.90 | 0.05 | 17.85 | 294.32 |
| Conv125 | 64 | 256 | 0.63 | 0.05 | 0.90 | 0.05 | 0.90 | 0.05 | 2.41 | 83.68 |
| Conv126 | 64 | 576 | 0.71 | 0.05 | 0.97 | 0.05 | 0.97 | 0.05 | 1.37 | 83.68 |
| Conv127 | 256 | 64 | 0.65 | 0.05 | 0.90 | 0.05 | 0.90 | 0.05 | 22.35 | 294.32 |
| Conv128 | 64 | 256 | 0.67 | 0.05 | 0.90 | 0.05 | 0.90 | 0.05 | 3.24 | 83.68 |

| Conv129 | 64 | 576 | 0.70 | 0.05 | 0.97 | 0.05 | 0.97 | 0.05 | 1.04 | 83.68 |
| Conv130 | 256 | 64 | 0.61 | 0.05 | 0.90 | 0.05 | 0.90 | 0.05 | 23.91 | 294.32 |
| Conv131 | 64 | 256 | 0.68 | 0.05 | 0.90 | 0.05 | 0.90 | 0.05 | 2.08 | 83.68 |
| Conv132 | 64 | 576 | 0.81 | 0.05 | 0.97 | 0.05 | 0.97 | 0.05 | 0.98 | 83.68 |
| Conv133 | 256 | 64 | 0.80 | 0.05 | 0.90 | 0.05 | 0.90 | 0.05 | 19.13 | 294.32 |
| Conv134 | 64 | 256 | 0.68 | 0.05 | 0.90 | 0.05 | 0.90 | 0.05 | 2.21 | 83.68 |
| Conv135 | 64 | 576 | 0.81 | 0.05 | 0.97 | 0.05 | 0.97 | 0.05 | 0.9 | 83.68 |
| Conv136 | 256 | 64 | 0.68 | 0.05 | 0.90 | 0.05 | 0.90 | 0.05 | 19.27 | 294.32 |
| Conv137 | 64 | 256 | 0.68 | 0.05 | 0.90 | 0.05 | 0.90 | 0.05 | 2.7 | 83.68 |
| Conv138 | 64 | 576 | 0.83 | 0.05 | 0.97 | 0.05 | 0.97 | 0.05 | 1.1 | 83.68 |
| Conv139 | 256 | 64 | 0.69 | 0.05 | 0.90 | 0.05 | 0.90 | 0.05 | 22.22 | 294.32 |
| Conv140 | 64 | 256 | 0.56 | 0.05 | 0.90 | 0.05 | 0.90 | 0.05 | 1.93 | 83.68 |
| Conv141 | 64 | 576 | 0.84 | 0.05 | 0.97 | 0.05 | 0.97 | 0.05 | 0.87 | 83.68 |
| Conv142 | 256 | 64 | 0.65 | 0.05 | 0.90 | 0.05 | 0.90 | 0.05 | 18.57 | 294.32 |
| Conv143 | 64 | 256 | 0.67 | 0.05 | 0.90 | 0.05 | 0.90 | 0.05 | 2.23 | 83.68 |
| Conv144 | 64 | 576 | 0.83 | 0.05 | 0.97 | 0.05 | 0.97 | 0.05 | 0.94 | 83.68 |
| Conv145 | 256 | 64 | 0.63 | 0.05 | 0.90 | 0.05 | 0.90 | 0.05 | 21.58 | 294.32 |
| Conv146 | 64 | 256 | 0.65 | 0.05 | 0.90 | 0.05 | 0.90 | 0.05 | 2.96 | 83.68 |
| Conv147 | 64 | 576 | 0.84 | 0.05 | 0.97 | 0.05 | 0.97 | 0.05 | 1.1 | 83.68 |
| Conv148 | 256 | 64 | 0.66 | 0.05 | 0.90 | 0.05 | 0.90 | 0.05 | 27.04 | 294.32 |
| Conv149 | 64 | 256 | 0.61 | 0.05 | 0.90 | 0.05 | 0.90 | 0.05 | 3.3 | 83.68 |
| Conv150 | 64 | 576 | 0.83 | 0.05 | 0.97 | 0.05 | 0.97 | 0.05 | 0.85 | 83.68 |
| Conv151 | 256 | 64 | 0.67 | 0.05 | 0.90 | 0.05 | 0.90 | 0.05 | 23.12 | 294.32 |
| Conv152 | 64 | 256 | 0.59 | 0.05 | 0.90 | 0.05 | 0.90 | 0.05 | 2.56 | 83.68 |
| Conv153 | 64 | 576 | 0.83 | 0.05 | 0.97 | 0.05 | 0.97 | 0.05 | 0.68 | 83.68 |
| Conv154 | 256 | 64 | 0.59 | 0.05 | 0.90 | 0.05 | 0.90 | 0.05 | 30.21 | 294.32 |
| Conv155 | 64 | 256 | 0.62 | 0.05 | 0.90 | 0.05 | 0.90 | 0.05 | 2.84 | 83.68 |
| Conv156 | 64 | 576 | 0.85 | 0.05 | 0.97 | 0.05 | 0.97 | 0.05 | 0.85 | 83.68 |
| Conv157 | 256 | 64 | 0.65 | 0.05 | 0.90 | 0.05 | 0.90 | 0.05 | 32.78 | 294.32 |
| Conv158 | 64 | 256 | 0.55 | 0.05 | 0.90 | 0.05 | 0.90 | 0.05 | 2.77 | 83.68 |
| Conv159 | 64 | 576 | 0.86 | 0.05 | 0.97 | 0.05 | 0.97 | 0.05 | 0.88 | 83.68 |
| Conv160 | 256 | 64 | 0.57 | 0.05 | 0.90 | 0.05 | 0.90 | 0.05 | 37.46 | 294.32 |
| Conv161 | 64 | 256 | 0.54 | 0.05 | 0.90 | 0.05 | 0.90 | 0.05 | 3.55 | 83.68 |
| Conv162 | 64 | 576 | 0.85 | 0.05 | 0.97 | 0.05 | 0.97 | 0.05 | 0.82 | 83.68 |
| Conv163 | 256 | 64 | 0.56 | 0.05 | 0.90 | 0.05 | 0.90 | 0.05 | 48.35 | 294.32 |
| **Passing rate** | - | - | 100.0% | | 100.0% | | 100.0% | | 100.00% | |

## P.3 REGULARIZATION

config:

```
https://github.com/bearpaw/pytorch-classification.
https://github.com/LeungSamWai/Drop-Activation
https://github.com/uoguelph-mlrg/Cutout
https://github.com/clovaai/CutMix-PyTorch
https://github.com/DeepVoltaire/AutoAugment
```

Table 20: WRN28-10 Cifar100 $\ell_1$

| Layer | Number | dim | Gaussian | | Mean_Left | | Mean_Right | | Sigma | |
|-------|--------|-----|---------|---------|-----------|---------|------------|---------|-------|---------|
| | | | p-value | c-value | p-value | c-value | p-value | c-value | t-value | c-value |
| Conv1 | 16 | 27 | 0.05 | 0.05 | 0.58 | 0.05 | 0.58 | 0.05 | 98.95 | 26.3 |
| Conv2 | 160 | 144 | 0.69 | 0.05 | 0.94 | 0.05 | 0.94 | 0.05 | 8.19 | 190.52 |
| Conv3 | 160 | 1440 | 0.93 | 0.05 | 1.00 | 0.05 | 1.00 | 0.05 | 0.31 | 190.52 |

| Layer | Number | dim | Gaussian | | Mean_Left | | Mean_Right | | Sigma | |
|---|---|---|---|---|---|---|---|---|---|---|
| | | | p-value | c-value | p-value | c-value | p-value | c-value | t-value | c-value |
| Conv4 | 160 | 16 | 0.58 | 0.05 | 0.69 | 0.05 | 0.69 | 0.05 | 76.38 | 190.52 |
| Conv5 | 160 | 1440 | 0.92 | 0.05 | 1.00 | 0.05 | 1.00 | 0.05 | 0.2 | 190.52 |
| Conv6 | 160 | 1440 | 0.92 | 0.05 | 1.00 | 0.05 | 1.00 | 0.05 | 0.25 | 190.52 |
| Conv7 | 160 | 1440 | 0.91 | 0.05 | 1.00 | 0.05 | 1.00 | 0.05 | 0.06 | 190.52 |
| Conv8 | 160 | 1440 | 0.93 | 0.05 | 1.00 | 0.05 | 1.00 | 0.05 | 0.25 | 190.52 |
| Conv9 | 160 | 1440 | 0.92 | 0.05 | 1.00 | 0.05 | 1.00 | 0.05 | 0.03 | 190.52 |
| Conv10 | 160 | 1440 | 0.94 | 0.05 | 1.00 | 0.05 | 1.00 | 0.05 | 0.95 | 190.52 |
| Conv11 | 320 | 1440 | 0.94 | 0.05 | 1.00 | 0.05 | 1.00 | 0.05 | 0.12 | 362.72 |
| Conv12 | 320 | 2880 | 0.94 | 0.05 | 1.00 | 0.05 | 1.00 | 0.05 | 0.19 | 362.72 |
| Conv13 | 320 | 160 | 0.84 | 0.05 | 0.99 | 0.05 | 0.99 | 0.05 | 4.26 | 362.72 |
| Conv14 | 320 | 2880 | 0.93 | 0.05 | 1.00 | 0.05 | 1.00 | 0.05 | 0.1 | 362.72 |
| Conv15 | 320 | 2880 | 0.94 | 0.05 | 1.00 | 0.05 | 1.00 | 0.05 | 0.16 | 362.72 |
| Conv16 | 320 | 2880 | 0.94 | 0.05 | 1.00 | 0.05 | 1.00 | 0.05 | 0.04 | 362.72 |
| Conv17 | 320 | 2880 | 0.94 | 0.05 | 1.00 | 0.05 | 1.00 | 0.05 | 0.26 | 362.72 |
| Conv18 | 320 | 2880 | 0.93 | 0.05 | 1.00 | 0.05 | 1.00 | 0.05 | 0.06 | 362.72 |
| Conv19 | 320 | 2880 | 0.93 | 0.05 | 1.00 | 0.05 | 1.00 | 0.05 | 0.52 | 362.72 |
| Conv20 | 640 | 2880 | 0.93 | 0.05 | 1.00 | 0.05 | 1.00 | 0.05 | 0.05 | 699.96 |
| Conv21 | 640 | 5760 | 0.93 | 0.05 | 1.00 | 0.05 | 1.00 | 0.05 | 0.64 | 699.96 |
| Conv22 | 640 | 320 | 0.91 | 0.05 | 1.00 | 0.05 | 1.00 | 0.05 | 2.88 | 699.96 |
| Conv23 | 640 | 5760 | 0.94 | 0.05 | 1.00 | 0.05 | 1.00 | 0.05 | 0.1 | 699.96 |
| Conv24 | 640 | 5760 | 0.93 | 0.05 | 1.00 | 0.05 | 1.00 | 0.05 | 0.49 | 699.96 |
| Conv25 | 640 | 5760 | 0.93 | 0.05 | 1.00 | 0.05 | 1.00 | 0.05 | 0.05 | 699.96 |
| Conv26 | 640 | 5760 | 0.94 | 0.05 | 1.00 | 0.05 | 1.00 | 0.05 | 0.46 | 699.96 |
| Conv27 | 640 | 5760 | 0.93 | 0.05 | 1.00 | 0.05 | 1.00 | 0.05 | 0.06 | 699.96 |
| Conv28 | 640 | 5760 | 0.94 | 0.05 | 1.00 | 0.05 | 1.00 | 0.05 | 2.74 | 699.96 |
| **Passing rate** | - | - | 100.0% | | 100.0% | | 100.0% | | 96.43% | |

Table 21: WRN28-10 Cifar100 RReLU

| Layer | Number | dim | Gaussian | | Mean_Left | | Mean_Right | | Sigma | |
|---|---|---|---|---|---|---|---|---|---|---|
| | | | p-value | c-value | p-value | c-value | p-value | c-value | t-value | c-value |
| Conv1 | 16 | 27 | 0.43 | 0.05 | 0.58 | 0.05 | 0.58 | 0.05 | 40.44 | 26.3 |
| Conv2 | 160 | 144 | 0.93 | 0.05 | 0.94 | 0.05 | 0.94 | 0.05 | 2.71 | 190.52 |
| Conv3 | 160 | 1440 | 0.95 | 0.05 | 1.00 | 0.05 | 1.00 | 0.05 | 0.12 | 190.52 |
| Conv4 | 160 | 16 | 0.82 | 0.05 | 0.69 | 0.05 | 0.69 | 0.05 | 31.44 | 190.52 |
| Conv5 | 160 | 1440 | 0.94 | 0.05 | 1.00 | 0.05 | 1.00 | 0.05 | 0.06 | 190.52 |
| Conv6 | 160 | 1440 | 0.95 | 0.05 | 1.00 | 0.05 | 1.00 | 0.05 | 0.12 | 190.52 |
| Conv7 | 160 | 1440 | 0.95 | 0.05 | 1.00 | 0.05 | 1.00 | 0.05 | 0.02 | 190.52 |
| Conv8 | 160 | 1440 | 0.95 | 0.05 | 1.00 | 0.05 | 1.00 | 0.05 | 0.2 | 190.52 |
| Conv9 | 160 | 1440 | 0.95 | 0.05 | 1.00 | 0.05 | 1.00 | 0.05 | 0.02 | 190.52 |
| Conv10 | 160 | 1440 | 0.95 | 0.05 | 1.00 | 0.05 | 1.00 | 0.05 | 0.3 | 190.52 |
| Conv11 | 320 | 1440 | 0.95 | 0.05 | 1.00 | 0.05 | 1.00 | 0.05 | 0.09 | 362.72 |
| Conv12 | 320 | 2880 | 0.95 | 0.05 | 1.00 | 0.05 | 1.00 | 0.05 | 0.09 | 362.72 |
| Conv13 | 320 | 160 | 0.90 | 0.05 | 0.99 | 0.05 | 0.99 | 0.05 | 2.73 | 362.72 |
| Conv14 | 320 | 2880 | 0.94 | 0.05 | 1.00 | 0.05 | 1.00 | 0.05 | 0.04 | 362.72 |
| Conv15 | 320 | 2880 | 0.95 | 0.05 | 1.00 | 0.05 | 1.00 | 0.05 | 0.07 | 362.72 |
| Conv16 | 320 | 2880 | 0.95 | 0.05 | 1.00 | 0.05 | 1.00 | 0.05 | 0.03 | 362.72 |
| Conv17 | 320 | 2880 | 0.95 | 0.05 | 1.00 | 0.05 | 1.00 | 0.05 | 0.14 | 362.72 |
| Conv18 | 320 | 2880 | 0.94 | 0.05 | 1.00 | 0.05 | 1.00 | 0.05 | 0.04 | 362.72 |
| Conv19 | 320 | 2880 | 0.95 | 0.05 | 1.00 | 0.05 | 1.00 | 0.05 | 0.31 | 362.72 |
| Conv20 | 640 | 2880 | 0.93 | 0.05 | 1.00 | 0.05 | 1.00 | 0.05 | 0.05 | 699.96 |
| Conv21 | 640 | 5760 | 0.94 | 0.05 | 1.00 | 0.05 | 1.00 | 0.05 | 0.33 | 699.96 |
| Conv22 | 640 | 320 | 0.93 | 0.05 | 1.00 | 0.05 | 1.00 | 0.05 | 1.54 | 699.96 |
| Conv23 | 640 | 5760 | 0.94 | 0.05 | 1.00 | 0.05 | 1.00 | 0.05 | 0.07 | 699.96 |
| Conv24 | 640 | 5760 | 0.95 | 0.05 | 1.00 | 0.05 | 1.00 | 0.05 | 0.28 | 699.96 |

| Layer | Number | dim | p-value | c-value | p-value | c-value | p-value | c-value | t-value | c-value |
|-------|--------|-----|---------|---------|---------|---------|---------|---------|---------|---------|
| Conv25 | 640 | 5760 | 0.95 | 0.05 | 1.00 | 0.05 | 1.00 | 0.05 | 0.05 | 699.96 |
| Conv26 | 640 | 5760 | 0.95 | 0.05 | 1.00 | 0.05 | 1.00 | 0.05 | 0.35 | 699.96 |
| Conv27 | 640 | 5760 | 0.94 | 0.05 | 1.00 | 0.05 | 1.00 | 0.05 | 0.04 | 699.96 |
| Conv28 | 640 | 5760 | 0.94 | 0.05 | 1.00 | 0.05 | 1.00 | 0.05 | 1.23 | 699.96 |
| **Passing rate** | - | - | 100.0% | | 100.0% | | 100.0% | | 96.43% | |

Table 22: WRN28-10 Cifar100 Dropact

| Layer | Number | dim | Gaussian | | Mean_Left | | Mean_Right | | Sigma | |
|-------|--------|-----|----------|----------|-----------|----------|------------|----------|-------|----------|
| | | | p-value | c-value | p-value | c-value | p-value | c-value | t-value | c-value |
| Conv1 | 16 | 27 | 0.54 | 0.05 | 0.58 | 0.05 | 0.58 | 0.05 | 19.13 | 26.3 |
| Conv2 | 160 | 144 | 0.92 | 0.05 | 0.94 | 0.05 | 0.94 | 0.05 | 1.61 | 190.52 |
| Conv3 | 160 | 1440 | 0.95 | 0.05 | 1.00 | 0.05 | 1.00 | 0.05 | 0.11 | 190.52 |
| Conv4 | 160 | 16 | 0.85 | 0.05 | 0.69 | 0.05 | 0.69 | 0.05 | 17.31 | 190.52 |
| Conv5 | 160 | 1440 | 0.94 | 0.05 | 1.00 | 0.05 | 1.00 | 0.05 | 0.06 | 190.52 |
| Conv6 | 160 | 1440 | 0.95 | 0.05 | 1.00 | 0.05 | 1.00 | 0.05 | 0.11 | 190.52 |
| Conv7 | 160 | 1440 | 0.95 | 0.05 | 1.00 | 0.05 | 1.00 | 0.05 | 0.01 | 190.52 |
| Conv8 | 160 | 1440 | 0.95 | 0.05 | 1.00 | 0.05 | 1.00 | 0.05 | 0.14 | 190.52 |
| Conv9 | 160 | 1440 | 0.95 | 0.05 | 1.00 | 0.05 | 1.00 | 0.05 | 0.02 | 190.52 |
| Conv10 | 160 | 1440 | 0.95 | 0.05 | 1.00 | 0.05 | 1.00 | 0.05 | 0.38 | 190.52 |
| Conv11 | 320 | 1440 | 0.95 | 0.05 | 1.00 | 0.05 | 1.00 | 0.05 | 0.06 | 362.72 |
| Conv12 | 320 | 2880 | 0.95 | 0.05 | 1.00 | 0.05 | 1.00 | 0.05 | 0.1 | 362.72 |
| Conv13 | 320 | 160 | 0.90 | 0.05 | 0.99 | 0.05 | 0.99 | 0.05 | 0.92 | 362.72 |
| Conv14 | 320 | 2880 | 0.95 | 0.05 | 1.00 | 0.05 | 1.00 | 0.05 | 0.03 | 362.72 |
| Conv15 | 320 | 2880 | 0.95 | 0.05 | 1.00 | 0.05 | 1.00 | 0.05 | 0.08 | 362.72 |
| Conv16 | 320 | 2880 | 0.94 | 0.05 | 1.00 | 0.05 | 1.00 | 0.05 | 0.03 | 362.72 |
| Conv17 | 320 | 2880 | 0.95 | 0.05 | 1.00 | 0.05 | 1.00 | 0.05 | 0.08 | 362.72 |
| Conv18 | 320 | 2880 | 0.94 | 0.05 | 1.00 | 0.05 | 1.00 | 0.05 | 0.04 | 362.72 |
| Conv19 | 320 | 2880 | 0.95 | 0.05 | 1.00 | 0.05 | 1.00 | 0.05 | 0.23 | 362.72 |
| Conv20 | 640 | 2880 | 0.95 | 0.05 | 1.00 | 0.05 | 1.00 | 0.05 | 0.05 | 699.96 |
| Conv21 | 640 | 5760 | 0.95 | 0.05 | 1.00 | 0.05 | 1.00 | 0.05 | 0.61 | 699.96 |
| Conv22 | 640 | 320 | 0.94 | 0.05 | 1.00 | 0.05 | 1.00 | 0.05 | 1.53 | 699.96 |
| Conv23 | 640 | 5760 | 0.95 | 0.05 | 1.00 | 0.05 | 1.00 | 0.05 | 0.06 | 699.96 |
| Conv24 | 640 | 5760 | 0.95 | 0.05 | 1.00 | 0.05 | 1.00 | 0.05 | 0.43 | 699.96 |
| Conv25 | 640 | 5760 | 0.95 | 0.05 | 1.00 | 0.05 | 1.00 | 0.05 | 0.05 | 699.96 |
| Conv26 | 640 | 5760 | 0.96 | 0.05 | 1.00 | 0.05 | 1.00 | 0.05 | 0.28 | 699.96 |
| Conv27 | 640 | 5760 | 0.95 | 0.05 | 1.00 | 0.05 | 1.00 | 0.05 | 0.04 | 699.96 |
| Conv28 | 640 | 5760 | 0.96 | 0.05 | 1.00 | 0.05 | 1.00 | 0.05 | 0.36 | 699.96 |
| **Passing rate** | - | - | 100.0% | | 100.0% | | 100.0% | | 100.0% | |

Table 23: WRN28-10 Cifar100 Autoaugment

| Layer | Number | dim | Gaussian | | Mean_Left | | Mean_Right | | Sigma | |
|-------|--------|-----|----------|----------|-----------|----------|------------|----------|-------|----------|
| | | | p-value | c-value | p-value | c-value | p-value | c-value | t-value | c-value |
| Conv1 | 16 | 27 | 0.05 | 0.05 | 0.58 | 0.05 | 0.58 | 0.05 | 100.18 | 26.3 |
| Conv2 | 160 | 144 | 0.69 | 0.05 | 0.94 | 0.05 | 0.94 | 0.05 | 8.3 | 190.52 |
| Conv3 | 160 | 1440 | 0.93 | 0.05 | 1.00 | 0.05 | 1.00 | 0.05 | 0.31 | 190.52 |
| Conv4 | 160 | 16 | 0.57 | 0.05 | 0.69 | 0.05 | 0.69 | 0.05 | 77.32 | 190.52 |
| Conv5 | 160 | 1440 | 0.92 | 0.05 | 1.00 | 0.05 | 1.00 | 0.05 | 0.2 | 190.52 |
| Conv6 | 160 | 1440 | 0.92 | 0.05 | 1.00 | 0.05 | 1.00 | 0.05 | 0.25 | 190.52 |
| Conv7 | 160 | 1440 | 0.91 | 0.05 | 1.00 | 0.05 | 1.00 | 0.05 | 0.06 | 190.52 |

| Conv8 | 160 | 1440 | 0.93 | 0.05 | 1.00 | 0.05 | 1.00 | 0.05 | 0.26 | 190.52 |
| Conv9 | 160 | 1440 | 0.92 | 0.05 | 1.00 | 0.05 | 1.00 | 0.05 | 0.03 | 190.52 |
| Conv10 | 160 | 1440 | 0.94 | 0.05 | 1.00 | 0.05 | 1.00 | 0.05 | 0.97 | 190.52 |
| Conv11 | 320 | 1440 | 0.94 | 0.05 | 1.00 | 0.05 | 1.00 | 0.05 | 0.12 | 362.72 |
| Conv12 | 320 | 2880 | 0.93 | 0.05 | 1.00 | 0.05 | 1.00 | 0.05 | 0.19 | 362.72 |
| Conv13 | 320 | 160 | 0.84 | 0.05 | 0.99 | 0.05 | 0.99 | 0.05 | 4.31 | 362.72 |
| Conv14 | 320 | 2880 | 0.93 | 0.05 | 1.00 | 0.05 | 1.00 | 0.05 | 0.1 | 362.72 |
| Conv15 | 320 | 2880 | 0.94 | 0.05 | 1.00 | 0.05 | 1.00 | 0.05 | 0.16 | 362.72 |
| Conv16 | 320 | 2880 | 0.94 | 0.05 | 1.00 | 0.05 | 1.00 | 0.05 | 0.04 | 362.72 |
| Conv17 | 320 | 2880 | 0.94 | 0.05 | 1.00 | 0.05 | 1.00 | 0.05 | 0.26 | 362.72 |
| Conv18 | 320 | 2880 | 0.93 | 0.05 | 1.00 | 0.05 | 1.00 | 0.05 | 0.06 | 362.72 |
| Conv19 | 320 | 2880 | 0.93 | 0.05 | 1.00 | 0.05 | 1.00 | 0.05 | 0.52 | 362.72 |
| Conv20 | 640 | 2880 | 0.93 | 0.05 | 1.00 | 0.05 | 1.00 | 0.05 | 0.05 | 699.96 |
| Conv21 | 640 | 5760 | 0.93 | 0.05 | 1.00 | 0.05 | 1.00 | 0.05 | 0.65 | 699.96 |
| Conv22 | 640 | 320 | 0.91 | 0.05 | 1.00 | 0.05 | 1.00 | 0.05 | 2.92 | 699.96 |
| Conv23 | 640 | 5760 | 0.94 | 0.05 | 1.00 | 0.05 | 1.00 | 0.05 | 0.1 | 699.96 |
| Conv24 | 640 | 5760 | 0.93 | 0.05 | 1.00 | 0.05 | 1.00 | 0.05 | 0.5 | 699.96 |
| Conv25 | 640 | 5760 | 0.93 | 0.05 | 1.00 | 0.05 | 1.00 | 0.05 | 0.05 | 699.96 |
| Conv26 | 640 | 5760 | 0.94 | 0.05 | 1.00 | 0.05 | 1.00 | 0.05 | 0.47 | 699.96 |
| Conv27 | 640 | 5760 | 0.93 | 0.05 | 1.00 | 0.05 | 1.00 | 0.05 | 0.06 | 699.96 |
| Conv28 | 640 | 5760 | 0.94 | 0.05 | 1.00 | 0.05 | 1.00 | 0.05 | 2.77 | 699.96 |
| **Passing rate** | - | - | 100.0% | | 100.0% | | 100.0% | | 96.43% | |

Table 24: WRN28-10 Cifar100 Cutout

| Layer | Number | dim | Gaussian | | Mean_Left | | Mean_Right | | Sigma | |
|---|---|---|---|---|---|---|---|---|---|---|
| | | | p-value | c-value | p-value | c-value | p-value | c-value | t-value | c-value |
| Conv1 | 16 | 27 | 0.31 | 0.05 | 0.58 | 0.05 | 0.58 | 0.05 | 49.46 | 26.3 |
| Conv2 | 160 | 144 | 0.88 | 0.05 | 0.94 | 0.05 | 0.94 | 0.05 | 5.66 | 190.52 |
| Conv3 | 160 | 1440 | 0.94 | 0.05 | 1.00 | 0.05 | 1.00 | 0.05 | 0.28 | 190.52 |
| Conv4 | 160 | 16 | 0.81 | 0.05 | 0.69 | 0.05 | 0.69 | 0.05 | 42.43 | 190.52 |
| Conv5 | 160 | 1440 | 0.93 | 0.05 | 1.00 | 0.05 | 1.00 | 0.05 | 0.18 | 190.52 |
| Conv6 | 160 | 1440 | 0.93 | 0.05 | 1.00 | 0.05 | 1.00 | 0.05 | 0.23 | 190.52 |
| Conv7 | 160 | 1440 | 0.94 | 0.05 | 1.00 | 0.05 | 1.00 | 0.05 | 0.02 | 190.52 |
| Conv8 | 160 | 1440 | 0.94 | 0.05 | 1.00 | 0.05 | 1.00 | 0.05 | 0.22 | 190.52 |
| Conv9 | 160 | 1440 | 0.94 | 0.05 | 1.00 | 0.05 | 1.00 | 0.05 | 0.06 | 190.52 |
| Conv10 | 160 | 1440 | 0.93 | 0.05 | 1.00 | 0.05 | 1.00 | 0.05 | 0.83 | 190.52 |
| Conv11 | 320 | 1440 | 0.94 | 0.05 | 1.00 | 0.05 | 1.00 | 0.05 | 0.1 | 362.72 |
| Conv12 | 320 | 2880 | 0.94 | 0.05 | 1.00 | 0.05 | 1.00 | 0.05 | 0.18 | 362.72 |
| Conv13 | 320 | 160 | 0.89 | 0.05 | 0.99 | 0.05 | 0.99 | 0.05 | 1.71 | 362.72 |
| Conv14 | 320 | 2880 | 0.94 | 0.05 | 1.00 | 0.05 | 1.00 | 0.05 | 0.05 | 362.72 |
| Conv15 | 320 | 2880 | 0.95 | 0.05 | 1.00 | 0.05 | 1.00 | 0.05 | 0.12 | 362.72 |
| Conv16 | 320 | 2880 | 0.94 | 0.05 | 1.00 | 0.05 | 1.00 | 0.05 | 0.06 | 362.72 |
| Conv17 | 320 | 2880 | 0.94 | 0.05 | 1.00 | 0.05 | 1.00 | 0.05 | 0.16 | 362.72 |
| Conv18 | 320 | 2880 | 0.94 | 0.05 | 1.00 | 0.05 | 1.00 | 0.05 | 0.07 | 362.72 |
| Conv19 | 320 | 2880 | 0.95 | 0.05 | 1.00 | 0.05 | 1.00 | 0.05 | 0.4 | 362.72 |
| Conv20 | 640 | 2880 | 0.93 | 0.05 | 1.00 | 0.05 | 1.00 | 0.05 | 0.05 | 699.96 |
| Conv21 | 640 | 5760 | 0.94 | 0.05 | 1.00 | 0.05 | 1.00 | 0.05 | 0.75 | 699.96 |
| Conv22 | 640 | 320 | 0.93 | 0.05 | 1.00 | 0.05 | 1.00 | 0.05 | 2.01 | 699.96 |
| Conv23 | 640 | 5760 | 0.95 | 0.05 | 1.00 | 0.05 | 1.00 | 0.05 | 0.12 | 699.96 |
| Conv24 | 640 | 5760 | 0.95 | 0.05 | 1.00 | 0.05 | 1.00 | 0.05 | 0.54 | 699.96 |
| Conv25 | 640 | 5760 | 0.95 | 0.05 | 1.00 | 0.05 | 1.00 | 0.05 | 0.1 | 699.96 |
| Conv26 | 640 | 5760 | 0.95 | 0.05 | 1.00 | 0.05 | 1.00 | 0.05 | 0.55 | 699.96 |
| Conv27 | 640 | 5760 | 0.95 | 0.05 | 1.00 | 0.05 | 1.00 | 0.05 | 0.09 | 699.96 |
| Conv28 | 640 | 5760 | 0.96 | 0.05 | 1.00 | 0.05 | 1.00 | 0.05 | 0.63 | 699.96 |

| Passing rate | - | - | 100.0% | 100.0% | 100.0% | 96.43% |

Table 25: WRN28-10 Cifar100 Cutmix

| Layer | Number | dim | Gaussian | | Mean_Left | | Mean_Right | | Sigma | |
|---|---|---|---|---|---|---|---|---|---|---|
| | | | p-value | c-value | p-value | c-value | p-value | c-value | t-value | c-value |
| Conv1 | 16 | 27 | 0.54 | 0.05 | 0.58 | 0.05 | 0.58 | 0.05 | 21.26 | 26.3 |
| Conv2 | 160 | 144 | 0.91 | 0.05 | 0.94 | 0.05 | 0.94 | 0.05 | 2.25 | 190.52 |
| Conv3 | 160 | 1440 | 0.95 | 0.05 | 1.00 | 0.05 | 1.00 | 0.05 | 0.16 | 190.52 |
| Conv4 | 160 | 16 | 0.86 | 0.05 | 0.69 | 0.05 | 0.69 | 0.05 | 19.09 | 190.52 |
| Conv5 | 160 | 1440 | 0.93 | 0.05 | 1.00 | 0.05 | 1.00 | 0.05 | 0.08 | 190.52 |
| Conv6 | 160 | 1440 | 0.94 | 0.05 | 1.00 | 0.05 | 1.00 | 0.05 | 0.14 | 190.52 |
| Conv7 | 160 | 1440 | 0.95 | 0.05 | 1.00 | 0.05 | 1.00 | 0.05 | 0.02 | 190.52 |
| Conv8 | 160 | 1440 | 0.95 | 0.05 | 1.00 | 0.05 | 1.00 | 0.05 | 0.14 | 190.52 |
| Conv9 | 160 | 1440 | 0.95 | 0.05 | 1.00 | 0.05 | 1.00 | 0.05 | 0.03 | 190.52 |
| Conv10 | 160 | 1440 | 0.95 | 0.05 | 1.00 | 0.05 | 1.00 | 0.05 | 0.29 | 190.52 |
| Conv11 | 320 | 1440 | 0.95 | 0.05 | 1.00 | 0.05 | 1.00 | 0.05 | 0.07 | 362.72 |
| Conv12 | 320 | 2880 | 0.95 | 0.05 | 1.00 | 0.05 | 1.00 | 0.05 | 0.12 | 362.72 |
| Conv13 | 320 | 160 | 0.92 | 0.05 | 0.99 | 0.05 | 0.99 | 0.05 | 1.09 | 362.72 |
| Conv14 | 320 | 2880 | 0.95 | 0.05 | 1.00 | 0.05 | 1.00 | 0.05 | 0.04 | 362.72 |
| Conv15 | 320 | 2880 | 0.95 | 0.05 | 1.00 | 0.05 | 1.00 | 0.05 | 0.08 | 362.72 |
| Conv16 | 320 | 2880 | 0.95 | 0.05 | 1.00 | 0.05 | 1.00 | 0.05 | 0.04 | 362.72 |
| Conv17 | 320 | 2880 | 0.95 | 0.05 | 1.00 | 0.05 | 1.00 | 0.05 | 0.09 | 362.72 |
| Conv18 | 320 | 2880 | 0.95 | 0.05 | 1.00 | 0.05 | 1.00 | 0.05 | 0.05 | 362.72 |
| Conv19 | 320 | 2880 | 0.95 | 0.05 | 1.00 | 0.05 | 1.00 | 0.05 | 0.26 | 362.72 |
| Conv20 | 640 | 2880 | 0.94 | 0.05 | 1.00 | 0.05 | 1.00 | 0.05 | 0.05 | 699.96 |
| Conv21 | 640 | 5760 | 0.95 | 0.05 | 1.00 | 0.05 | 1.00 | 0.05 | 0.85 | 699.96 |
| Conv22 | 640 | 320 | 0.94 | 0.05 | 1.00 | 0.05 | 1.00 | 0.05 | 1.8 | 699.96 |
| Conv23 | 640 | 5760 | 0.95 | 0.05 | 1.00 | 0.05 | 1.00 | 0.05 | 0.09 | 699.96 |
| Conv24 | 640 | 5760 | 0.95 | 0.05 | 1.00 | 0.05 | 1.00 | 0.05 | 0.51 | 699.96 |
| Conv25 | 640 | 5760 | 0.95 | 0.05 | 1.00 | 0.05 | 1.00 | 0.05 | 0.07 | 699.96 |
| Conv26 | 640 | 5760 | 0.96 | 0.05 | 1.00 | 0.05 | 1.00 | 0.05 | 0.22 | 699.96 |
| Conv27 | 640 | 5760 | 0.96 | 0.05 | 1.00 | 0.05 | 1.00 | 0.05 | 0.04 | 699.96 |
| Conv28 | 640 | 5760 | 0.96 | 0.05 | 1.00 | 0.05 | 1.00 | 0.05 | 0.24 | 699.96 |
| **Passing rate** | - | - | 100.0% | | 100.0% | | 100.0% | | 100.0% | |

## P.4 ATTENTION MECHANISM

config:

https://github.com/moskomule/senet.pytorch

https://github.com/gbup-group/DIANet

https://github.com/EvgenyKashin/SRMnet

https://github.com/luuuyi/CBAM.PyTorch

https://github.com/gbup-group/IEBN

https://github.com/implus/PytorchInsight

Table 26: SENet (ResNet164) Cifar100

| Layer | Number | dim | Gaussian | Mean_Left | Mean_Right | Sigma |
|---|---|---|---|---|---|---|

| | | | p-value | c-value | p-value | c-value | p-value | c-value | t-value | c-value |
|---|---|---|---|---|---|---|---|---|---|---|
| Conv1 | 16 | 27 | 0.02 | 0.05 | 0.58 | 0.05 | 0.58 | 0.05 | 181.46 | 26.3 |
| Conv2 | 16 | 16 | 0.54 | 0.05 | 0.56 | 0.05 | 0.56 | 0.05 | 6.12 | 26.3 |
| Conv3 | 16 | 144 | 0.86 | 0.05 | 0.68 | 0.05 | 0.68 | 0.05 | 0.42 | 26.3 |
| Conv4 | 64 | 16 | 0.87 | 0.05 | 0.63 | 0.05 | 0.63 | 0.05 | 11.54 | 83.68 |
| Conv5 | 16 | 64 | 0.82 | 0.05 | 0.63 | 0.05 | 0.63 | 0.05 | 0.89 | 26.3 |
| Conv6 | 16 | 144 | 0.89 | 0.05 | 0.68 | 0.05 | 0.68 | 0.05 | 0.37 | 26.3 |
| Conv7 | 64 | 16 | 0.80 | 0.05 | 0.63 | 0.05 | 0.63 | 0.05 | 10.59 | 83.68 |
| Conv8 | 16 | 64 | 0.67 | 0.05 | 0.63 | 0.05 | 0.63 | 0.05 | 1.54 | 26.3 |
| Conv9 | 16 | 144 | 0.82 | 0.05 | 0.68 | 0.05 | 0.68 | 0.05 | 1.34 | 26.3 |
| Conv10 | 64 | 16 | 0.64 | 0.05 | 0.63 | 0.05 | 0.63 | 0.05 | 11.73 | 83.68 |
| Conv11 | 16 | 64 | 0.81 | 0.05 | 0.63 | 0.05 | 0.63 | 0.05 | 1.01 | 26.3 |
| Conv12 | 16 | 144 | 0.84 | 0.05 | 0.68 | 0.05 | 0.68 | 0.05 | 1.63 | 26.3 |
| Conv13 | 64 | 16 | 0.81 | 0.05 | 0.63 | 0.05 | 0.63 | 0.05 | 9.92 | 83.68 |
| Conv14 | 16 | 64 | 0.43 | 0.05 | 0.63 | 0.05 | 0.63 | 0.05 | 3.99 | 26.3 |
| Conv15 | 16 | 144 | 0.81 | 0.05 | 0.68 | 0.05 | 0.68 | 0.05 | 0.86 | 26.3 |
| Conv16 | 64 | 16 | 0.81 | 0.05 | 0.63 | 0.05 | 0.63 | 0.05 | 11.13 | 83.68 |
| Conv17 | 16 | 64 | 0.67 | 0.05 | 0.63 | 0.05 | 0.63 | 0.05 | 0.68 | 26.3 |
| Conv18 | 16 | 144 | 0.84 | 0.05 | 0.68 | 0.05 | 0.68 | 0.05 | 0.61 | 26.3 |
| Conv19 | 64 | 16 | 0.86 | 0.05 | 0.63 | 0.05 | 0.63 | 0.05 | 12.02 | 83.68 |
| Conv20 | 16 | 64 | 0.87 | 0.05 | 0.63 | 0.05 | 0.63 | 0.05 | 0.65 | 26.3 |
| Conv21 | 16 | 144 | 0.84 | 0.05 | 0.68 | 0.05 | 0.68 | 0.05 | 1.02 | 26.3 |
| Conv22 | 64 | 16 | 0.85 | 0.05 | 0.63 | 0.05 | 0.63 | 0.05 | 7.73 | 83.68 |
| Conv23 | 16 | 64 | 0.69 | 0.05 | 0.63 | 0.05 | 0.63 | 0.05 | 1.84 | 26.3 |
| Conv24 | 16 | 144 | 0.89 | 0.05 | 0.68 | 0.05 | 0.68 | 0.05 | 1.07 | 26.3 |
| Conv25 | 64 | 16 | 0.85 | 0.05 | 0.63 | 0.05 | 0.63 | 0.05 | 7.12 | 83.68 |
| Conv26 | 16 | 64 | 0.69 | 0.05 | 0.63 | 0.05 | 0.63 | 0.05 | 1.85 | 26.3 |
| Conv27 | 16 | 144 | 0.82 | 0.05 | 0.68 | 0.05 | 0.68 | 0.05 | 0.77 | 26.3 |
| Conv28 | 64 | 16 | 0.86 | 0.05 | 0.63 | 0.05 | 0.63 | 0.05 | 14.12 | 83.68 |
| Conv29 | 16 | 64 | 0.64 | 0.05 | 0.63 | 0.05 | 0.63 | 0.05 | 3.44 | 26.3 |
| Conv30 | 16 | 144 | 0.81 | 0.05 | 0.68 | 0.05 | 0.68 | 0.05 | 0.81 | 26.3 |
| Conv31 | 64 | 16 | 0.81 | 0.05 | 0.63 | 0.05 | 0.63 | 0.05 | 17.04 | 83.68 |
| Conv32 | 16 | 64 | 0.68 | 0.05 | 0.63 | 0.05 | 0.63 | 0.05 | 2.15 | 26.3 |
| Conv33 | 16 | 144 | 0.67 | 0.05 | 0.68 | 0.05 | 0.68 | 0.05 | 0.81 | 26.3 |
| Conv34 | 64 | 16 | 0.71 | 0.05 | 0.63 | 0.05 | 0.63 | 0.05 | 23.95 | 83.68 |
| Conv35 | 16 | 64 | 0.69 | 0.05 | 0.63 | 0.05 | 0.63 | 0.05 | 1.03 | 26.3 |
| Conv36 | 16 | 144 | 0.89 | 0.05 | 0.68 | 0.05 | 0.68 | 0.05 | 0.1 | 26.3 |
| Conv37 | 64 | 16 | 0.86 | 0.05 | 0.63 | 0.05 | 0.63 | 0.05 | 8.09 | 83.68 |
| Conv38 | 16 | 64 | 0.68 | 0.05 | 0.63 | 0.05 | 0.63 | 0.05 | 3.55 | 26.3 |
| Conv39 | 16 | 144 | 0.83 | 0.05 | 0.68 | 0.05 | 0.68 | 0.05 | 0.71 | 26.3 |
| Conv40 | 64 | 16 | 0.84 | 0.05 | 0.63 | 0.05 | 0.63 | 0.05 | 11.45 | 83.68 |
| Conv41 | 16 | 64 | 0.81 | 0.05 | 0.63 | 0.05 | 0.63 | 0.05 | 1.84 | 26.3 |
| Conv42 | 16 | 144 | 0.82 | 0.05 | 0.68 | 0.05 | 0.68 | 0.05 | 1.1 | 26.3 |
| Conv43 | 64 | 16 | 0.83 | 0.05 | 0.63 | 0.05 | 0.63 | 0.05 | 10.49 | 83.68 |
| Conv44 | 16 | 64 | 0.81 | 0.05 | 0.63 | 0.05 | 0.63 | 0.05 | 2.3 | 26.3 |
| Conv45 | 16 | 144 | 0.84 | 0.05 | 0.68 | 0.05 | 0.68 | 0.05 | 1.96 | 26.3 |
| Conv46 | 64 | 16 | 0.46 | 0.05 | 0.63 | 0.05 | 0.63 | 0.05 | 18.47 | 83.68 |
| Conv47 | 16 | 64 | 0.67 | 0.05 | 0.63 | 0.05 | 0.63 | 0.05 | 2.86 | 26.3 |
| Conv48 | 16 | 144 | 0.82 | 0.05 | 0.68 | 0.05 | 0.68 | 0.05 | 0.84 | 26.3 |
| Conv49 | 64 | 16 | 0.64 | 0.05 | 0.63 | 0.05 | 0.63 | 0.05 | 8.9 | 83.68 |
| Conv50 | 16 | 64 | 0.82 | 0.05 | 0.63 | 0.05 | 0.63 | 0.05 | 3.4 | 26.3 |
| Conv51 | 16 | 144 | 0.84 | 0.05 | 0.68 | 0.05 | 0.68 | 0.05 | 0.92 | 26.3 |
| Conv52 | 64 | 16 | 0.71 | 0.05 | 0.63 | 0.05 | 0.63 | 0.05 | 14.29 | 83.68 |
| Conv53 | 16 | 64 | 0.47 | 0.05 | 0.63 | 0.05 | 0.63 | 0.05 | 5.93 | 26.3 |
| Conv54 | 16 | 144 | 0.86 | 0.05 | 0.68 | 0.05 | 0.68 | 0.05 | 0.92 | 26.3 |
| Conv55 | 64 | 16 | 0.64 | 0.05 | 0.63 | 0.05 | 0.63 | 0.05 | 16.37 | 83.68 |
| Conv56 | 32 | 64 | 0.51 | 0.05 | 0.67 | 0.05 | 0.67 | 0.05 | 3.25 | 46.19 |
| Conv57 | 32 | 288 | 0.88 | 0.05 | 0.83 | 0.05 | 0.83 | 0.05 | 0.24 | 46.19 |

| Conv58 | 128 | 32 | 0.80 | 0.05 | 0.74 | 0.05 | 0.74 | 0.05 | 14.02 | 155.4 |
|--------|-----|----|------|------|------|------|------|------|-------|-------|
| Conv59 | 32 | 128 | 0.90 | 0.05 | 0.74 | 0.05 | 0.74 | 0.05 | 0.44 | 46.19 |
| Conv60 | 32 | 288 | 0.88 | 0.05 | 0.83 | 0.05 | 0.83 | 0.05 | 0.37 | 46.19 |
| Conv61 | 128 | 32 | 0.88 | 0.05 | 0.74 | 0.05 | 0.74 | 0.05 | 5.11 | 155.4 |
| Conv62 | 32 | 128 | 0.85 | 0.05 | 0.74 | 0.05 | 0.74 | 0.05 | 0.72 | 46.19 |
| Conv63 | 32 | 288 | 0.88 | 0.05 | 0.83 | 0.05 | 0.83 | 0.05 | 0.5 | 46.19 |
| Conv64 | 128 | 32 | 0.85 | 0.05 | 0.74 | 0.05 | 0.74 | 0.05 | 5.3 | 155.4 |
| Conv65 | 32 | 128 | 0.82 | 0.05 | 0.74 | 0.05 | 0.74 | 0.05 | 1.19 | 46.19 |
| Conv66 | 32 | 288 | 0.81 | 0.05 | 0.83 | 0.05 | 0.83 | 0.05 | 1.05 | 46.19 |
| Conv67 | 128 | 32 | 0.81 | 0.05 | 0.74 | 0.05 | 0.74 | 0.05 | 8.62 | 155.4 |
| Conv68 | 32 | 128 | 0.87 | 0.05 | 0.74 | 0.05 | 0.74 | 0.05 | 0.41 | 46.19 |
| Conv69 | 32 | 288 | 0.89 | 0.05 | 0.83 | 0.05 | 0.83 | 0.05 | 0.73 | 46.19 |
| Conv70 | 128 | 32 | 0.85 | 0.05 | 0.74 | 0.05 | 0.74 | 0.05 | 5.21 | 155.4 |
| Conv71 | 32 | 128 | 0.89 | 0.05 | 0.74 | 0.05 | 0.74 | 0.05 | 0.38 | 46.19 |
| Conv72 | 32 | 288 | 0.88 | 0.05 | 0.83 | 0.05 | 0.83 | 0.05 | 0.5 | 46.19 |
| Conv73 | 128 | 32 | 0.88 | 0.05 | 0.74 | 0.05 | 0.74 | 0.05 | 4.26 | 155.4 |
| Conv74 | 32 | 128 | 0.83 | 0.05 | 0.74 | 0.05 | 0.74 | 0.05 | 0.85 | 46.19 |
| Conv75 | 32 | 288 | 0.87 | 0.05 | 0.83 | 0.05 | 0.83 | 0.05 | 0.41 | 46.19 |
| Conv76 | 128 | 32 | 0.68 | 0.05 | 0.74 | 0.05 | 0.74 | 0.05 | 6.07 | 155.4 |
| Conv77 | 32 | 128 | 0.85 | 0.05 | 0.74 | 0.05 | 0.74 | 0.05 | 0.52 | 46.19 |
| Conv78 | 32 | 288 | 0.88 | 0.05 | 0.83 | 0.05 | 0.83 | 0.05 | 0.44 | 46.19 |
| Conv79 | 128 | 32 | 0.84 | 0.05 | 0.74 | 0.05 | 0.74 | 0.05 | 4.39 | 155.4 |
| Conv80 | 32 | 128 | 0.82 | 0.05 | 0.74 | 0.05 | 0.74 | 0.05 | 0.83 | 46.19 |
| Conv81 | 32 | 288 | 0.90 | 0.05 | 0.83 | 0.05 | 0.83 | 0.05 | 0.75 | 46.19 |
| Conv82 | 128 | 32 | 0.88 | 0.05 | 0.74 | 0.05 | 0.74 | 0.05 | 5.93 | 155.4 |
| Conv83 | 32 | 128 | 0.68 | 0.05 | 0.74 | 0.05 | 0.74 | 0.05 | 2.33 | 46.19 |
| Conv84 | 32 | 288 | 0.87 | 0.05 | 0.83 | 0.05 | 0.83 | 0.05 | 0.57 | 46.19 |
| Conv85 | 128 | 32 | 0.84 | 0.05 | 0.74 | 0.05 | 0.74 | 0.05 | 8.22 | 155.4 |
| Conv86 | 32 | 128 | 0.82 | 0.05 | 0.74 | 0.05 | 0.74 | 0.05 | 1.0 | 46.19 |
| Conv87 | 32 | 288 | 0.81 | 0.05 | 0.83 | 0.05 | 0.83 | 0.05 | 0.43 | 46.19 |
| Conv88 | 128 | 32 | 0.89 | 0.05 | 0.74 | 0.05 | 0.74 | 0.05 | 5.04 | 155.4 |
| Conv89 | 32 | 128 | 0.86 | 0.05 | 0.74 | 0.05 | 0.74 | 0.05 | 1.08 | 46.19 |
| Conv90 | 32 | 288 | 0.91 | 0.05 | 0.83 | 0.05 | 0.83 | 0.05 | 0.34 | 46.19 |
| Conv91 | 128 | 32 | 0.87 | 0.05 | 0.74 | 0.05 | 0.74 | 0.05 | 5.44 | 155.4 |
| Conv92 | 32 | 128 | 0.86 | 0.05 | 0.74 | 0.05 | 0.74 | 0.05 | 0.93 | 46.19 |
| Conv93 | 32 | 288 | 0.89 | 0.05 | 0.83 | 0.05 | 0.83 | 0.05 | 0.24 | 46.19 |
| Conv94 | 128 | 32 | 0.86 | 0.05 | 0.74 | 0.05 | 0.74 | 0.05 | 5.26 | 155.4 |
| Conv95 | 32 | 128 | 0.83 | 0.05 | 0.74 | 0.05 | 0.74 | 0.05 | 0.92 | 46.19 |
| Conv96 | 32 | 288 | 0.90 | 0.05 | 0.83 | 0.05 | 0.83 | 0.05 | 0.19 | 46.19 |
| Conv97 | 128 | 32 | 0.87 | 0.05 | 0.74 | 0.05 | 0.74 | 0.05 | 5.4 | 155.4 |
| Conv98 | 32 | 128 | 0.87 | 0.05 | 0.74 | 0.05 | 0.74 | 0.05 | 0.92 | 46.19 |
| Conv99 | 32 | 288 | 0.86 | 0.05 | 0.83 | 0.05 | 0.83 | 0.05 | 0.4 | 46.19 |
| Conv100 | 128 | 32 | 0.88 | 0.05 | 0.74 | 0.05 | 0.74 | 0.05 | 5.19 | 155.4 |
| Conv101 | 32 | 128 | 0.85 | 0.05 | 0.74 | 0.05 | 0.74 | 0.05 | 1.2 | 46.19 |
| Conv102 | 32 | 288 | 0.89 | 0.05 | 0.83 | 0.05 | 0.83 | 0.05 | 0.37 | 46.19 |
| Conv103 | 128 | 32 | 0.70 | 0.05 | 0.74 | 0.05 | 0.74 | 0.05 | 6.58 | 155.4 |
| Conv104 | 32 | 128 | 0.85 | 0.05 | 0.74 | 0.05 | 0.74 | 0.05 | 1.03 | 46.19 |
| Conv105 | 32 | 288 | 0.89 | 0.05 | 0.83 | 0.05 | 0.83 | 0.05 | 0.15 | 46.19 |
| Conv106 | 128 | 32 | 0.87 | 0.05 | 0.74 | 0.05 | 0.74 | 0.05 | 6.01 | 155.4 |
| Conv107 | 32 | 128 | 0.86 | 0.05 | 0.74 | 0.05 | 0.74 | 0.05 | 0.98 | 46.19 |
| Conv108 | 32 | 288 | 0.88 | 0.05 | 0.83 | 0.05 | 0.83 | 0.05 | 0.19 | 46.19 |
| Conv109 | 128 | 32 | 0.85 | 0.05 | 0.74 | 0.05 | 0.74 | 0.05 | 6.87 | 155.4 |
| Conv110 | 64 | 128 | 0.61 | 0.05 | 0.82 | 0.05 | 0.82 | 0.05 | 2.17 | 83.68 |
| Conv111 | 64 | 576 | 0.89 | 0.05 | 0.97 | 0.05 | 0.97 | 0.05 | 0.18 | 83.68 |
| Conv112 | 256 | 64 | 0.83 | 0.05 | 0.90 | 0.05 | 0.90 | 0.05 | 12.64 | 294.32 |
| Conv113 | 64 | 256 | 0.86 | 0.05 | 0.90 | 0.05 | 0.90 | 0.05 | 0.68 | 83.68 |
| Conv114 | 64 | 576 | 0.89 | 0.05 | 0.97 | 0.05 | 0.97 | 0.05 | 0.32 | 83.68 |
| Conv115 | 256 | 64 | 0.84 | 0.05 | 0.90 | 0.05 | 0.90 | 0.05 | 6.51 | 294.32 |
| Conv116 | 64 | 256 | 0.70 | 0.05 | 0.90 | 0.05 | 0.90 | 0.05 | 0.67 | 83.68 |

| Layer | Number | dim | | | | | | | | |
|-------|--------|-----|------|------|------|------|------|------|------|------|
| Conv117 | 64 | 576 | 0.89 | 0.05 | 0.97 | 0.05 | 0.97 | 0.05 | 0.46 | 83.68 |
| Conv118 | 256 | 64 | 0.86 | 0.05 | 0.90 | 0.05 | 0.90 | 0.05 | 7.04 | 294.32 |
| Conv119 | 64 | 256 | 0.86 | 0.05 | 0.90 | 0.05 | 0.90 | 0.05 | 0.57 | 83.68 |
| Conv120 | 64 | 576 | 0.85 | 0.05 | 0.97 | 0.05 | 0.97 | 0.05 | 0.35 | 83.68 |
| Conv121 | 256 | 64 | 0.81 | 0.05 | 0.90 | 0.05 | 0.90 | 0.05 | 6.19 | 294.32 |
| Conv122 | 64 | 256 | 0.82 | 0.05 | 0.90 | 0.05 | 0.90 | 0.05 | 0.36 | 83.68 |
| Conv123 | 64 | 576 | 0.90 | 0.05 | 0.97 | 0.05 | 0.97 | 0.05 | 0.35 | 83.68 |
| Conv124 | 256 | 64 | 0.86 | 0.05 | 0.90 | 0.05 | 0.90 | 0.05 | 6.66 | 294.32 |
| Conv125 | 64 | 256 | 0.87 | 0.05 | 0.90 | 0.05 | 0.90 | 0.05 | 0.4 | 83.68 |
| Conv126 | 64 | 576 | 0.90 | 0.05 | 0.97 | 0.05 | 0.97 | 0.05 | 0.29 | 83.68 |
| Conv127 | 256 | 64 | 0.85 | 0.05 | 0.90 | 0.05 | 0.90 | 0.05 | 7.23 | 294.32 |
| Conv128 | 64 | 256 | 0.86 | 0.05 | 0.90 | 0.05 | 0.90 | 0.05 | 0.47 | 83.68 |
| Conv129 | 64 | 576 | 0.88 | 0.05 | 0.97 | 0.05 | 0.97 | 0.05 | 0.25 | 83.68 |
| Conv130 | 256 | 64 | 0.86 | 0.05 | 0.90 | 0.05 | 0.90 | 0.05 | 6.13 | 294.32 |
| Conv131 | 64 | 256 | 0.86 | 0.05 | 0.90 | 0.05 | 0.90 | 0.05 | 0.56 | 83.68 |
| Conv132 | 64 | 576 | 0.89 | 0.05 | 0.97 | 0.05 | 0.97 | 0.05 | 0.3 | 83.68 |
| Conv133 | 256 | 64 | 0.86 | 0.05 | 0.90 | 0.05 | 0.90 | 0.05 | 6.56 | 294.32 |
| Conv134 | 64 | 256 | 0.85 | 0.05 | 0.90 | 0.05 | 0.90 | 0.05 | 0.54 | 83.68 |
| Conv135 | 64 | 576 | 0.88 | 0.05 | 0.97 | 0.05 | 0.97 | 0.05 | 0.25 | 83.68 |
| Conv136 | 256 | 64 | 0.86 | 0.05 | 0.90 | 0.05 | 0.90 | 0.05 | 6.57 | 294.32 |
| Conv137 | 64 | 256 | 0.86 | 0.05 | 0.90 | 0.05 | 0.90 | 0.05 | 0.69 | 83.68 |
| Conv138 | 64 | 576 | 0.88 | 0.05 | 0.97 | 0.05 | 0.97 | 0.05 | 0.2 | 83.68 |
| Conv139 | 256 | 64 | 0.83 | 0.05 | 0.90 | 0.05 | 0.90 | 0.05 | 5.95 | 294.32 |
| Conv140 | 64 | 256 | 0.84 | 0.05 | 0.90 | 0.05 | 0.90 | 0.05 | 0.4 | 83.68 |
| Conv141 | 64 | 576 | 0.90 | 0.05 | 0.97 | 0.05 | 0.97 | 0.05 | 0.24 | 83.68 |
| Conv142 | 256 | 64 | 0.83 | 0.05 | 0.90 | 0.05 | 0.90 | 0.05 | 6.64 | 294.32 |
| Conv143 | 64 | 256 | 0.87 | 0.05 | 0.90 | 0.05 | 0.90 | 0.05 | 0.65 | 83.68 |
| Conv144 | 64 | 576 | 0.91 | 0.05 | 0.97 | 0.05 | 0.97 | 0.05 | 0.19 | 83.68 |
| Conv145 | 256 | 64 | 0.83 | 0.05 | 0.90 | 0.05 | 0.90 | 0.05 | 7.35 | 294.32 |
| Conv146 | 64 | 256 | 0.69 | 0.05 | 0.90 | 0.05 | 0.90 | 0.05 | 0.66 | 83.68 |
| Conv147 | 64 | 576 | 0.87 | 0.05 | 0.97 | 0.05 | 0.97 | 0.05 | 0.14 | 83.68 |
| Conv148 | 256 | 64 | 0.84 | 0.05 | 0.90 | 0.05 | 0.90 | 0.05 | 7.95 | 294.32 |
| Conv149 | 64 | 256 | 0.85 | 0.05 | 0.90 | 0.05 | 0.90 | 0.05 | 0.59 | 83.68 |
| Conv150 | 64 | 576 | 0.90 | 0.05 | 0.97 | 0.05 | 0.97 | 0.05 | 0.18 | 83.68 |
| Conv151 | 256 | 64 | 0.82 | 0.05 | 0.90 | 0.05 | 0.90 | 0.05 | 6.92 | 294.32 |
| Conv152 | 64 | 256 | 0.82 | 0.05 | 0.90 | 0.05 | 0.90 | 0.05 | 0.42 | 83.68 |
| Conv153 | 64 | 576 | 0.91 | 0.05 | 0.97 | 0.05 | 0.97 | 0.05 | 0.14 | 83.68 |
| Conv154 | 256 | 64 | 0.85 | 0.05 | 0.90 | 0.05 | 0.90 | 0.05 | 7.74 | 294.32 |
| Conv155 | 64 | 256 | 0.82 | 0.05 | 0.90 | 0.05 | 0.90 | 0.05 | 0.58 | 83.68 |
| Conv156 | 64 | 576 | 0.91 | 0.05 | 0.97 | 0.05 | 0.97 | 0.05 | 0.16 | 83.68 |
| Conv157 | 256 | 64 | 0.84 | 0.05 | 0.90 | 0.05 | 0.90 | 0.05 | 7.89 | 294.32 |
| Conv158 | 64 | 256 | 0.84 | 0.05 | 0.90 | 0.05 | 0.90 | 0.05 | 0.47 | 83.68 |
| Conv159 | 64 | 576 | 0.91 | 0.05 | 0.97 | 0.05 | 0.97 | 0.05 | 0.19 | 83.68 |
| Conv160 | 256 | 64 | 0.85 | 0.05 | 0.90 | 0.05 | 0.90 | 0.05 | 8.11 | 294.32 |
| Conv161 | 64 | 256 | 0.81 | 0.05 | 0.90 | 0.05 | 0.90 | 0.05 | 0.83 | 83.68 |
| Conv162 | 64 | 576 | 0.91 | 0.05 | 0.97 | 0.05 | 0.97 | 0.05 | 0.15 | 83.68 |
| Conv163 | 256 | 64 | 0.82 | 0.05 | 0.90 | 0.05 | 0.90 | 0.05 | 9.08 | 294.32 |
| **Passing rate** | - | - | 99.39% | | 100.0% | | 100.0% | | 99.39% | |

Table 27: DIANet (ResNet164) Cifar100

| Layer | Number | dim | Gaussian | | Mean_Left | | Mean_Right | | Sigma | |
|-------|--------|-----|----------|---------|-----------|---------|------------|---------|---------|---------|
| | | | p-value | c-value | p-value | c-value | p-value | c-value | t-value | c-value |
| Conv1 | 16 | 27 | 0.02 | 0.05 | 0.58 | 0.05 | 0.58 | 0.05 | 218.97 | 26.3 |
| Conv2 | 16 | 16 | 0.41 | 0.05 | 0.56 | 0.05 | 0.56 | 0.05 | 16.01 | 26.3 |

| | | | | | | | | | | |
|---|---|---|---|---|---|---|---|---|---|---|
| Conv3 | 16 | 144 | 0.54 | 0.05 | 0.68 | 0.05 | 0.68 | 0.05 | 2.13 | 26.3 |
| Conv4 | 64 | 16 | 0.83 | 0.05 | 0.63 | 0.05 | 0.63 | 0.05 | 22.48 | 83.68 |
| Conv5 | 16 | 64 | 0.82 | 0.05 | 0.63 | 0.05 | 0.63 | 0.05 | 1.21 | 26.3 |
| Conv6 | 16 | 144 | 0.88 | 0.05 | 0.68 | 0.05 | 0.68 | 0.05 | 0.52 | 26.3 |
| Conv7 | 64 | 16 | 0.83 | 0.05 | 0.63 | 0.05 | 0.63 | 0.05 | 10.96 | 83.68 |
| Conv8 | 16 | 64 | 0.81 | 0.05 | 0.63 | 0.05 | 0.63 | 0.05 | 1.12 | 26.3 |
| Conv9 | 16 | 144 | 0.88 | 0.05 | 0.68 | 0.05 | 0.68 | 0.05 | 0.65 | 26.3 |
| Conv10 | 64 | 16 | 0.86 | 0.05 | 0.63 | 0.05 | 0.63 | 0.05 | 6.11 | 83.68 |
| Conv11 | 16 | 64 | 0.84 | 0.05 | 0.63 | 0.05 | 0.63 | 0.05 | 1.32 | 26.3 |
| Conv12 | 16 | 144 | 0.87 | 0.05 | 0.68 | 0.05 | 0.68 | 0.05 | 0.4 | 26.3 |
| Conv13 | 64 | 16 | 0.86 | 0.05 | 0.63 | 0.05 | 0.63 | 0.05 | 10.91 | 83.68 |
| Conv14 | 16 | 64 | 0.81 | 0.05 | 0.63 | 0.05 | 0.63 | 0.05 | 2.18 | 26.3 |
| Conv15 | 16 | 144 | 0.69 | 0.05 | 0.68 | 0.05 | 0.68 | 0.05 | 1.12 | 26.3 |
| Conv16 | 64 | 16 | 0.82 | 0.05 | 0.63 | 0.05 | 0.63 | 0.05 | 24.69 | 83.68 |
| Conv17 | 16 | 64 | 0.83 | 0.05 | 0.63 | 0.05 | 0.63 | 0.05 | 0.87 | 26.3 |
| Conv18 | 16 | 144 | 0.85 | 0.05 | 0.68 | 0.05 | 0.68 | 0.05 | 0.32 | 26.3 |
| Conv19 | 64 | 16 | 0.85 | 0.05 | 0.63 | 0.05 | 0.63 | 0.05 | 10.98 | 83.68 |
| Conv20 | 16 | 64 | 0.62 | 0.05 | 0.63 | 0.05 | 0.63 | 0.05 | 1.4 | 26.3 |
| Conv21 | 16 | 144 | 0.89 | 0.05 | 0.68 | 0.05 | 0.68 | 0.05 | 0.41 | 26.3 |
| Conv22 | 64 | 16 | 0.70 | 0.05 | 0.63 | 0.05 | 0.63 | 0.05 | 15.49 | 83.68 |
| Conv23 | 16 | 64 | 0.84 | 0.05 | 0.63 | 0.05 | 0.63 | 0.05 | 1.03 | 26.3 |
| Conv24 | 16 | 144 | 0.90 | 0.05 | 0.68 | 0.05 | 0.68 | 0.05 | 0.77 | 26.3 |
| Conv25 | 64 | 16 | 0.83 | 0.05 | 0.63 | 0.05 | 0.63 | 0.05 | 10.43 | 83.68 |
| Conv26 | 16 | 64 | 0.82 | 0.05 | 0.63 | 0.05 | 0.63 | 0.05 | 1.59 | 26.3 |
| Conv27 | 16 | 144 | 0.86 | 0.05 | 0.68 | 0.05 | 0.68 | 0.05 | 0.47 | 26.3 |
| Conv28 | 64 | 16 | 0.71 | 0.05 | 0.63 | 0.05 | 0.63 | 0.05 | 11.21 | 83.68 |
| Conv29 | 16 | 64 | 0.65 | 0.05 | 0.63 | 0.05 | 0.63 | 0.05 | 2.22 | 26.3 |
| Conv30 | 16 | 144 | 0.81 | 0.05 | 0.68 | 0.05 | 0.68 | 0.05 | 0.78 | 26.3 |
| Conv31 | 64 | 16 | 0.86 | 0.05 | 0.63 | 0.05 | 0.63 | 0.05 | 9.33 | 83.68 |
| Conv32 | 16 | 64 | 0.62 | 0.05 | 0.63 | 0.05 | 0.63 | 0.05 | 2.48 | 26.3 |
| Conv33 | 16 | 144 | 0.85 | 0.05 | 0.68 | 0.05 | 0.68 | 0.05 | 1.23 | 26.3 |
| Conv34 | 64 | 16 | 0.81 | 0.05 | 0.63 | 0.05 | 0.63 | 0.05 | 10.78 | 83.68 |
| Conv35 | 16 | 64 | 0.59 | 0.05 | 0.63 | 0.05 | 0.63 | 0.05 | 3.47 | 26.3 |
| Conv36 | 16 | 144 | 0.84 | 0.05 | 0.68 | 0.05 | 0.68 | 0.05 | 0.78 | 26.3 |
| Conv37 | 64 | 16 | 0.83 | 0.05 | 0.63 | 0.05 | 0.63 | 0.05 | 17.75 | 83.68 |
| Conv38 | 16 | 64 | 0.87 | 0.05 | 0.63 | 0.05 | 0.63 | 0.05 | 0.81 | 26.3 |
| Conv39 | 16 | 144 | 0.85 | 0.05 | 0.68 | 0.05 | 0.68 | 0.05 | 0.57 | 26.3 |
| Conv40 | 64 | 16 | 0.84 | 0.05 | 0.63 | 0.05 | 0.63 | 0.05 | 12.64 | 83.68 |
| Conv41 | 16 | 64 | 0.67 | 0.05 | 0.63 | 0.05 | 0.63 | 0.05 | 1.94 | 26.3 |
| Conv42 | 16 | 144 | 0.87 | 0.05 | 0.68 | 0.05 | 0.68 | 0.05 | 0.61 | 26.3 |
| Conv43 | 64 | 16 | 0.69 | 0.05 | 0.63 | 0.05 | 0.63 | 0.05 | 10.1 | 83.68 |
| Conv44 | 16 | 64 | 0.63 | 0.05 | 0.63 | 0.05 | 0.63 | 0.05 | 3.55 | 26.3 |
| Conv45 | 16 | 144 | 0.70 | 0.05 | 0.68 | 0.05 | 0.68 | 0.05 | 1.23 | 26.3 |
| Conv46 | 64 | 16 | 0.61 | 0.05 | 0.63 | 0.05 | 0.63 | 0.05 | 49.74 | 83.68 |
| Conv47 | 16 | 64 | 0.88 | 0.05 | 0.63 | 0.05 | 0.63 | 0.05 | 1.3 | 26.3 |
| Conv48 | 16 | 144 | 0.87 | 0.05 | 0.68 | 0.05 | 0.68 | 0.05 | 0.34 | 26.3 |
| Conv49 | 64 | 16 | 0.85 | 0.05 | 0.63 | 0.05 | 0.63 | 0.05 | 7.77 | 83.68 |
| Conv50 | 16 | 64 | 0.81 | 0.05 | 0.63 | 0.05 | 0.63 | 0.05 | 1.92 | 26.3 |
| Conv51 | 16 | 144 | 0.87 | 0.05 | 0.68 | 0.05 | 0.68 | 0.05 | 0.67 | 26.3 |
| Conv52 | 64 | 16 | 0.82 | 0.05 | 0.63 | 0.05 | 0.63 | 0.05 | 13.42 | 83.68 |
| Conv53 | 16 | 64 | 0.69 | 0.05 | 0.63 | 0.05 | 0.63 | 0.05 | 3.6 | 26.3 |
| Conv54 | 16 | 144 | 0.81 | 0.05 | 0.68 | 0.05 | 0.68 | 0.05 | 0.83 | 26.3 |
| Conv55 | 64 | 16 | 0.70 | 0.05 | 0.63 | 0.05 | 0.63 | 0.05 | 24.26 | 83.68 |
| Conv56 | 32 | 64 | 0.62 | 0.05 | 0.67 | 0.05 | 0.67 | 0.05 | 6.86 | 46.19 |
| Conv57 | 32 | 288 | 0.85 | 0.05 | 0.83 | 0.05 | 0.83 | 0.05 | 0.39 | 46.19 |
| Conv58 | 128 | 32 | 0.61 | 0.05 | 0.74 | 0.05 | 0.74 | 0.05 | 26.14 | 155.4 |
| Conv59 | 32 | 128 | 0.87 | 0.05 | 0.74 | 0.05 | 0.74 | 0.05 | 0.65 | 46.19 |
| Conv60 | 32 | 288 | 0.87 | 0.05 | 0.83 | 0.05 | 0.83 | 0.05 | 0.44 | 46.19 |
| Conv61 | 128 | 32 | 0.85 | 0.05 | 0.74 | 0.05 | 0.74 | 0.05 | 10.89 | 155.4 |

| | | | | | | | | | | | |
|---|---|---|---|---|---|---|---|---|---|---|---|
| Conv62 | 32 | 128 | 0.86 | 0.05 | 0.74 | 0.05 | 0.74 | 0.05 | 0.76 | 46.19 |
| Conv63 | 32 | 288 | 0.89 | 0.05 | 0.83 | 0.05 | 0.83 | 0.05 | 0.62 | 46.19 |
| Conv64 | 128 | 32 | 0.84 | 0.05 | 0.74 | 0.05 | 0.74 | 0.05 | 9.18 | 155.4 |
| Conv65 | 32 | 128 | 0.87 | 0.05 | 0.74 | 0.05 | 0.74 | 0.05 | 0.53 | 46.19 |
| Conv66 | 32 | 288 | 0.88 | 0.05 | 0.83 | 0.05 | 0.83 | 0.05 | 0.56 | 46.19 |
| Conv67 | 128 | 32 | 0.84 | 0.05 | 0.74 | 0.05 | 0.74 | 0.05 | 6.94 | 155.4 |
| Conv68 | 32 | 128 | 0.89 | 0.05 | 0.74 | 0.05 | 0.74 | 0.05 | 0.53 | 46.19 |
| Conv69 | 32 | 288 | 0.88 | 0.05 | 0.83 | 0.05 | 0.83 | 0.05 | 0.45 | 46.19 |
| Conv70 | 128 | 32 | 0.65 | 0.05 | 0.74 | 0.05 | 0.74 | 0.05 | 8.26 | 155.4 |
| Conv71 | 32 | 128 | 0.88 | 0.05 | 0.74 | 0.05 | 0.74 | 0.05 | 0.62 | 46.19 |
| Conv72 | 32 | 288 | 0.70 | 0.05 | 0.83 | 0.05 | 0.83 | 0.05 | 0.41 | 46.19 |
| Conv73 | 128 | 32 | 0.83 | 0.05 | 0.74 | 0.05 | 0.74 | 0.05 | 6.53 | 155.4 |
| Conv74 | 32 | 128 | 0.69 | 0.05 | 0.74 | 0.05 | 0.74 | 0.05 | 0.77 | 46.19 |
| Conv75 | 32 | 288 | 0.91 | 0.05 | 0.83 | 0.05 | 0.83 | 0.05 | 0.27 | 46.19 |
| Conv76 | 128 | 32 | 0.85 | 0.05 | 0.74 | 0.05 | 0.74 | 0.05 | 9.43 | 155.4 |
| Conv77 | 32 | 128 | 0.68 | 0.05 | 0.74 | 0.05 | 0.74 | 0.05 | 1.07 | 46.19 |
| Conv78 | 32 | 288 | 0.89 | 0.05 | 0.83 | 0.05 | 0.83 | 0.05 | 0.36 | 46.19 |
| Conv79 | 128 | 32 | 0.84 | 0.05 | 0.74 | 0.05 | 0.74 | 0.05 | 10.11 | 155.4 |
| Conv80 | 32 | 128 | 0.83 | 0.05 | 0.74 | 0.05 | 0.74 | 0.05 | 0.59 | 46.19 |
| Conv81 | 32 | 288 | 0.89 | 0.05 | 0.83 | 0.05 | 0.83 | 0.05 | 0.29 | 46.19 |
| Conv82 | 128 | 32 | 0.89 | 0.05 | 0.74 | 0.05 | 0.74 | 0.05 | 6.81 | 155.4 |
| Conv83 | 32 | 128 | 0.82 | 0.05 | 0.74 | 0.05 | 0.74 | 0.05 | 1.28 | 46.19 |
| Conv84 | 32 | 288 | 0.88 | 0.05 | 0.83 | 0.05 | 0.83 | 0.05 | 0.63 | 46.19 |
| Conv85 | 128 | 32 | 0.84 | 0.05 | 0.74 | 0.05 | 0.74 | 0.05 | 12.87 | 155.4 |
| Conv86 | 32 | 128 | 0.69 | 0.05 | 0.74 | 0.05 | 0.74 | 0.05 | 0.58 | 46.19 |
| Conv87 | 32 | 288 | 0.89 | 0.05 | 0.83 | 0.05 | 0.83 | 0.05 | 0.28 | 46.19 |
| Conv88 | 128 | 32 | 0.86 | 0.05 | 0.74 | 0.05 | 0.74 | 0.05 | 8.96 | 155.4 |
| Conv89 | 32 | 128 | 0.84 | 0.05 | 0.74 | 0.05 | 0.74 | 0.05 | 1.59 | 46.19 |
| Conv90 | 32 | 288 | 0.89 | 0.05 | 0.83 | 0.05 | 0.83 | 0.05 | 0.1 | 46.19 |
| Conv91 | 128 | 32 | 0.87 | 0.05 | 0.74 | 0.05 | 0.74 | 0.05 | 9.08 | 155.4 |
| Conv92 | 32 | 128 | 0.84 | 0.05 | 0.74 | 0.05 | 0.74 | 0.05 | 1.02 | 46.19 |
| Conv93 | 32 | 288 | 0.90 | 0.05 | 0.83 | 0.05 | 0.83 | 0.05 | 0.09 | 46.19 |
| Conv94 | 128 | 32 | 0.85 | 0.05 | 0.74 | 0.05 | 0.74 | 0.05 | 8.1 | 155.4 |
| Conv95 | 32 | 128 | 0.85 | 0.05 | 0.74 | 0.05 | 0.74 | 0.05 | 0.96 | 46.19 |
| Conv96 | 32 | 288 | 0.89 | 0.05 | 0.83 | 0.05 | 0.83 | 0.05 | 0.19 | 46.19 |
| Conv97 | 128 | 32 | 0.84 | 0.05 | 0.74 | 0.05 | 0.74 | 0.05 | 8.75 | 155.4 |
| Conv98 | 32 | 128 | 0.83 | 0.05 | 0.74 | 0.05 | 0.74 | 0.05 | 1.11 | 46.19 |
| Conv99 | 32 | 288 | 0.90 | 0.05 | 0.83 | 0.05 | 0.83 | 0.05 | 0.22 | 46.19 |
| Conv100 | 128 | 32 | 0.87 | 0.05 | 0.74 | 0.05 | 0.74 | 0.05 | 8.92 | 155.4 |
| Conv101 | 32 | 128 | 0.85 | 0.05 | 0.74 | 0.05 | 0.74 | 0.05 | 1.12 | 46.19 |
| Conv102 | 32 | 288 | 0.88 | 0.05 | 0.83 | 0.05 | 0.83 | 0.05 | 0.2 | 46.19 |
| Conv103 | 128 | 32 | 0.86 | 0.05 | 0.74 | 0.05 | 0.74 | 0.05 | 8.21 | 155.4 |
| Conv104 | 32 | 128 | 0.84 | 0.05 | 0.74 | 0.05 | 0.74 | 0.05 | 1.03 | 46.19 |
| Conv105 | 32 | 288 | 0.84 | 0.05 | 0.83 | 0.05 | 0.83 | 0.05 | 0.31 | 46.19 |
| Conv106 | 128 | 32 | 0.86 | 0.05 | 0.74 | 0.05 | 0.74 | 0.05 | 7.93 | 155.4 |
| Conv107 | 32 | 128 | 0.70 | 0.05 | 0.74 | 0.05 | 0.74 | 0.05 | 1.65 | 46.19 |
| Conv108 | 32 | 288 | 0.89 | 0.05 | 0.83 | 0.05 | 0.83 | 0.05 | 0.32 | 46.19 |
| Conv109 | 128 | 32 | 0.86 | 0.05 | 0.74 | 0.05 | 0.74 | 0.05 | 8.7 | 155.4 |
| Conv110 | 64 | 128 | 0.65 | 0.05 | 0.82 | 0.05 | 0.82 | 0.05 | 1.96 | 83.68 |
| Conv111 | 64 | 576 | 0.87 | 0.05 | 0.97 | 0.05 | 0.97 | 0.05 | 0.18 | 83.68 |
| Conv112 | 256 | 64 | 0.70 | 0.05 | 0.90 | 0.05 | 0.90 | 0.05 | 15.81 | 294.32 |
| Conv113 | 64 | 256 | 0.60 | 0.05 | 0.90 | 0.05 | 0.90 | 0.05 | 0.85 | 83.68 |
| Conv114 | 64 | 576 | 0.88 | 0.05 | 0.97 | 0.05 | 0.97 | 0.05 | 0.41 | 83.68 |
| Conv115 | 256 | 64 | 0.82 | 0.05 | 0.90 | 0.05 | 0.90 | 0.05 | 6.26 | 294.32 |
| Conv116 | 64 | 256 | 0.82 | 0.05 | 0.90 | 0.05 | 0.90 | 0.05 | 0.61 | 83.68 |
| Conv117 | 64 | 576 | 0.89 | 0.05 | 0.97 | 0.05 | 0.97 | 0.05 | 0.31 | 83.68 |
| Conv118 | 256 | 64 | 0.84 | 0.05 | 0.90 | 0.05 | 0.90 | 0.05 | 6.85 | 294.32 |
| Conv119 | 64 | 256 | 0.69 | 0.05 | 0.90 | 0.05 | 0.90 | 0.05 | 0.54 | 83.68 |
| Conv120 | 64 | 576 | 0.89 | 0.05 | 0.97 | 0.05 | 0.97 | 0.05 | 0.35 | 83.68 |

| Layer | | | | | | | | | |
|---|---|---|---|---|---|---|---|---|---|
| Conv121 | 256 | 64 | 0.82 | 0.05 | 0.90 | 0.05 | 0.90 | 0.05 | 6.53 | 294.32 |
| Conv122 | 64 | 256 | 0.85 | 0.05 | 0.90 | 0.05 | 0.90 | 0.05 | 0.43 | 83.68 |
| Conv123 | 64 | 576 | 0.89 | 0.05 | 0.97 | 0.05 | 0.97 | 0.05 | 0.39 | 83.68 |
| Conv124 | 256 | 64 | 0.66 | 0.05 | 0.90 | 0.05 | 0.90 | 0.05 | 6.93 | 294.32 |
| Conv125 | 64 | 256 | 0.67 | 0.05 | 0.90 | 0.05 | 0.90 | 0.05 | 0.6 | 83.68 |
| Conv126 | 64 | 576 | 0.88 | 0.05 | 0.97 | 0.05 | 0.97 | 0.05 | 0.25 | 83.68 |
| Conv127 | 256 | 64 | 0.71 | 0.05 | 0.90 | 0.05 | 0.90 | 0.05 | 8.9 | 294.32 |
| Conv128 | 64 | 256 | 0.85 | 0.05 | 0.90 | 0.05 | 0.90 | 0.05 | 0.47 | 83.68 |
| Conv129 | 64 | 576 | 0.88 | 0.05 | 0.97 | 0.05 | 0.97 | 0.05 | 0.18 | 83.68 |
| Conv130 | 256 | 64 | 0.83 | 0.05 | 0.90 | 0.05 | 0.90 | 0.05 | 6.85 | 294.32 |
| Conv131 | 64 | 256 | 0.86 | 0.05 | 0.90 | 0.05 | 0.90 | 0.05 | 0.47 | 83.68 |
| Conv132 | 64 | 576 | 0.88 | 0.05 | 0.97 | 0.05 | 0.97 | 0.05 | 0.17 | 83.68 |
| Conv133 | 256 | 64 | 0.86 | 0.05 | 0.90 | 0.05 | 0.90 | 0.05 | 5.22 | 294.32 |
| Conv134 | 64 | 256 | 0.71 | 0.05 | 0.90 | 0.05 | 0.90 | 0.05 | 0.4 | 83.68 |
| Conv135 | 64 | 576 | 0.89 | 0.05 | 0.97 | 0.05 | 0.97 | 0.05 | 0.22 | 83.68 |
| Conv136 | 256 | 64 | 0.70 | 0.05 | 0.90 | 0.05 | 0.90 | 0.05 | 7.07 | 294.32 |
| Conv137 | 64 | 256 | 0.84 | 0.05 | 0.90 | 0.05 | 0.90 | 0.05 | 0.39 | 83.68 |
| Conv138 | 64 | 576 | 0.87 | 0.05 | 0.97 | 0.05 | 0.97 | 0.05 | 0.22 | 83.68 |
| Conv139 | 256 | 64 | 0.85 | 0.05 | 0.90 | 0.05 | 0.90 | 0.05 | 5.24 | 294.32 |
| Conv140 | 64 | 256 | 0.84 | 0.05 | 0.90 | 0.05 | 0.90 | 0.05 | 0.55 | 83.68 |
| Conv141 | 64 | 576 | 0.89 | 0.05 | 0.97 | 0.05 | 0.97 | 0.05 | 0.21 | 83.68 |
| Conv142 | 256 | 64 | 0.83 | 0.05 | 0.90 | 0.05 | 0.90 | 0.05 | 7.08 | 294.32 |
| Conv143 | 64 | 256 | 0.84 | 0.05 | 0.90 | 0.05 | 0.90 | 0.05 | 0.36 | 83.68 |
| Conv144 | 64 | 576 | 0.87 | 0.05 | 0.97 | 0.05 | 0.97 | 0.05 | 0.26 | 83.68 |
| Conv145 | 256 | 64 | 0.82 | 0.05 | 0.90 | 0.05 | 0.90 | 0.05 | 8.28 | 294.32 |
| Conv146 | 64 | 256 | 0.83 | 0.05 | 0.90 | 0.05 | 0.90 | 0.05 | 0.42 | 83.68 |
| Conv147 | 64 | 576 | 0.87 | 0.05 | 0.97 | 0.05 | 0.97 | 0.05 | 0.27 | 83.68 |
| Conv148 | 256 | 64 | 0.83 | 0.05 | 0.90 | 0.05 | 0.90 | 0.05 | 8.01 | 294.32 |
| Conv149 | 64 | 256 | 0.69 | 0.05 | 0.90 | 0.05 | 0.90 | 0.05 | 0.68 | 83.68 |
| Conv150 | 64 | 576 | 0.90 | 0.05 | 0.97 | 0.05 | 0.97 | 0.05 | 0.17 | 83.68 |
| Conv151 | 256 | 64 | 0.82 | 0.05 | 0.90 | 0.05 | 0.90 | 0.05 | 7.8 | 294.32 |
| Conv152 | 64 | 256 | 0.80 | 0.05 | 0.90 | 0.05 | 0.90 | 0.05 | 0.39 | 83.68 |
| Conv153 | 64 | 576 | 0.89 | 0.05 | 0.97 | 0.05 | 0.97 | 0.05 | 0.11 | 83.68 |
| Conv154 | 256 | 64 | 0.81 | 0.05 | 0.90 | 0.05 | 0.90 | 0.05 | 8.1 | 294.32 |
| Conv155 | 64 | 256 | 0.80 | 0.05 | 0.90 | 0.05 | 0.90 | 0.05 | 0.55 | 83.68 |
| Conv156 | 64 | 576 | 0.89 | 0.05 | 0.97 | 0.05 | 0.97 | 0.05 | 0.11 | 83.68 |
| Conv157 | 256 | 64 | 0.82 | 0.05 | 0.90 | 0.05 | 0.90 | 0.05 | 8.92 | 294.32 |
| Conv158 | 64 | 256 | 0.69 | 0.05 | 0.90 | 0.05 | 0.90 | 0.05 | 0.61 | 83.68 |
| Conv159 | 64 | 576 | 0.90 | 0.05 | 0.97 | 0.05 | 0.97 | 0.05 | 0.1 | 83.68 |
| Conv160 | 256 | 64 | 0.70 | 0.05 | 0.90 | 0.05 | 0.90 | 0.05 | 10.08 | 294.32 |
| Conv161 | 64 | 256 | 0.82 | 0.05 | 0.90 | 0.05 | 0.90 | 0.05 | 0.44 | 83.68 |
| Conv162 | 64 | 576 | 0.91 | 0.05 | 0.97 | 0.05 | 0.97 | 0.05 | 0.09 | 83.68 |
| Conv163 | 256 | 64 | 0.81 | 0.05 | 0.90 | 0.05 | 0.90 | 0.05 | 11.0 | 294.32 |
| **Passing rate** | - | - | 99.39% | | 100.0% | | 100.0% | | 99.39% | |

Table 28: SRMNet (ResNet164) Cifar100

| Layer | Number | dim | Gaussian | | Mean_Left | | Mean_Right | | Sigma | |
|---|---|---|---|---|---|---|---|---|---|---|
| | | | p-value | c-value | p-value | c-value | p-value | c-value | t-value | c-value |
| Conv1 | 16 | 27 | 0.00 | 0.05 | 0.58 | 0.05 | 0.58 | 0.05 | 1623.9 | 26.3 |
| Conv2 | 16 | 16 | 0.35 | 0.05 | 0.56 | 0.05 | 0.56 | 0.05 | 33.36 | 26.3 |
| Conv3 | 16 | 144 | 0.28 | 0.05 | 0.68 | 0.05 | 0.68 | 0.05 | 21.42 | 26.3 |
| Conv4 | 64 | 16 | 0.31 | 0.05 | 0.63 | 0.05 | 0.63 | 0.05 | 156.52 | 83.68 |
| Conv5 | 16 | 64 | 0.54 | 0.05 | 0.63 | 0.05 | 0.63 | 0.05 | 9.2 | 26.3 |
| Conv6 | 16 | 144 | 0.43 | 0.05 | 0.68 | 0.05 | 0.68 | 0.05 | 17.06 | 26.3 |

| | | | | | | | | | | |
|---|---|---|---|---|---|---|---|---|---|---|
| Conv7 | 64 | 16 | 0.16 | 0.05 | 0.63 | 0.05 | 0.63 | 0.05 | 141.38 | 83.68 |
| Conv8 | 16 | 64 | 0.71 | 0.05 | 0.63 | 0.05 | 0.63 | 0.05 | 1.14 | 26.3 |
| Conv9 | 16 | 144 | 0.64 | 0.05 | 0.68 | 0.05 | 0.68 | 0.05 | 0.84 | 26.3 |
| Conv10 | 64 | 16 | 0.68 | 0.05 | 0.63 | 0.05 | 0.63 | 0.05 | 26.28 | 83.68 |
| Conv11 | 16 | 64 | 0.86 | 0.05 | 0.63 | 0.05 | 0.63 | 0.05 | 0.6 | 26.3 |
| Conv12 | 16 | 144 | 0.82 | 0.05 | 0.68 | 0.05 | 0.68 | 0.05 | 1.66 | 26.3 |
| Conv13 | 64 | 16 | 0.71 | 0.05 | 0.63 | 0.05 | 0.63 | 0.05 | 19.29 | 83.68 |
| Conv14 | 16 | 64 | 0.68 | 0.05 | 0.63 | 0.05 | 0.63 | 0.05 | 1.02 | 26.3 |
| Conv15 | 16 | 144 | 0.82 | 0.05 | 0.68 | 0.05 | 0.68 | 0.05 | 1.34 | 26.3 |
| Conv16 | 64 | 16 | 0.85 | 0.05 | 0.63 | 0.05 | 0.63 | 0.05 | 6.93 | 83.68 |
| Conv17 | 16 | 64 | 0.86 | 0.05 | 0.63 | 0.05 | 0.63 | 0.05 | 1.13 | 26.3 |
| Conv18 | 16 | 144 | 0.87 | 0.05 | 0.68 | 0.05 | 0.68 | 0.05 | 0.33 | 26.3 |
| Conv19 | 64 | 16 | 0.87 | 0.05 | 0.63 | 0.05 | 0.63 | 0.05 | 3.44 | 83.68 |
| Conv20 | 16 | 64 | 0.82 | 0.05 | 0.63 | 0.05 | 0.63 | 0.05 | 1.76 | 26.3 |
| Conv21 | 16 | 144 | 0.87 | 0.05 | 0.68 | 0.05 | 0.68 | 0.05 | 0.82 | 26.3 |
| Conv22 | 64 | 16 | 0.86 | 0.05 | 0.63 | 0.05 | 0.63 | 0.05 | 5.63 | 83.68 |
| Conv23 | 16 | 64 | 0.87 | 0.05 | 0.63 | 0.05 | 0.63 | 0.05 | 1.43 | 26.3 |
| Conv24 | 16 | 144 | 0.88 | 0.05 | 0.68 | 0.05 | 0.68 | 0.05 | 0.71 | 26.3 |
| Conv25 | 64 | 16 | 0.86 | 0.05 | 0.63 | 0.05 | 0.63 | 0.05 | 4.67 | 83.68 |
| Conv26 | 16 | 64 | 0.63 | 0.05 | 0.63 | 0.05 | 0.63 | 0.05 | 2.74 | 26.3 |
| Conv27 | 16 | 144 | 0.64 | 0.05 | 0.68 | 0.05 | 0.68 | 0.05 | 2.32 | 26.3 |
| Conv28 | 64 | 16 | 0.84 | 0.05 | 0.63 | 0.05 | 0.63 | 0.05 | 7.18 | 83.68 |
| Conv29 | 16 | 64 | 0.65 | 0.05 | 0.63 | 0.05 | 0.63 | 0.05 | 3.72 | 26.3 |
| Conv30 | 16 | 144 | 0.52 | 0.05 | 0.68 | 0.05 | 0.68 | 0.05 | 2.86 | 26.3 |
| Conv31 | 64 | 16 | 0.84 | 0.05 | 0.63 | 0.05 | 0.63 | 0.05 | 10.89 | 83.68 |
| Conv32 | 16 | 64 | 0.85 | 0.05 | 0.63 | 0.05 | 0.63 | 0.05 | 2.79 | 26.3 |
| Conv33 | 16 | 144 | 0.86 | 0.05 | 0.68 | 0.05 | 0.68 | 0.05 | 0.69 | 26.3 |
| Conv34 | 64 | 16 | 0.86 | 0.05 | 0.63 | 0.05 | 0.63 | 0.05 | 8.82 | 83.68 |
| Conv35 | 16 | 64 | 0.83 | 0.05 | 0.63 | 0.05 | 0.63 | 0.05 | 3.59 | 26.3 |
| Conv36 | 16 | 144 | 0.81 | 0.05 | 0.68 | 0.05 | 0.68 | 0.05 | 1.62 | 26.3 |
| Conv37 | 64 | 16 | 0.86 | 0.05 | 0.63 | 0.05 | 0.63 | 0.05 | 6.81 | 83.68 |
| Conv38 | 16 | 64 | 0.71 | 0.05 | 0.63 | 0.05 | 0.63 | 0.05 | 2.18 | 26.3 |
| Conv39 | 16 | 144 | 0.61 | 0.05 | 0.68 | 0.05 | 0.68 | 0.05 | 2.3 | 26.3 |
| Conv40 | 64 | 16 | 0.84 | 0.05 | 0.63 | 0.05 | 0.63 | 0.05 | 8.24 | 83.68 |
| Conv41 | 16 | 64 | 0.81 | 0.05 | 0.63 | 0.05 | 0.63 | 0.05 | 2.79 | 26.3 |
| Conv42 | 16 | 144 | 0.63 | 0.05 | 0.68 | 0.05 | 0.68 | 0.05 | 3.23 | 26.3 |
| Conv43 | 64 | 16 | 0.86 | 0.05 | 0.63 | 0.05 | 0.63 | 0.05 | 8.83 | 83.68 |
| Conv44 | 16 | 64 | 0.62 | 0.05 | 0.63 | 0.05 | 0.63 | 0.05 | 5.95 | 26.3 |
| Conv45 | 16 | 144 | 0.65 | 0.05 | 0.68 | 0.05 | 0.68 | 0.05 | 4.0 | 26.3 |
| Conv46 | 64 | 16 | 0.80 | 0.05 | 0.63 | 0.05 | 0.63 | 0.05 | 13.79 | 83.68 |
| Conv47 | 16 | 64 | 0.81 | 0.05 | 0.63 | 0.05 | 0.63 | 0.05 | 2.02 | 26.3 |
| Conv48 | 16 | 144 | 0.85 | 0.05 | 0.68 | 0.05 | 0.68 | 0.05 | 1.49 | 26.3 |
| Conv49 | 64 | 16 | 0.90 | 0.05 | 0.63 | 0.05 | 0.63 | 0.05 | 5.35 | 83.68 |
| Conv50 | 16 | 64 | 0.69 | 0.05 | 0.63 | 0.05 | 0.63 | 0.05 | 10.35 | 26.3 |
| Conv51 | 16 | 144 | 0.60 | 0.05 | 0.68 | 0.05 | 0.68 | 0.05 | 2.38 | 26.3 |
| Conv52 | 64 | 16 | 0.66 | 0.05 | 0.63 | 0.05 | 0.63 | 0.05 | 27.58 | 83.68 |
| Conv53 | 16 | 64 | 0.87 | 0.05 | 0.63 | 0.05 | 0.63 | 0.05 | 0.87 | 26.3 |
| Conv54 | 16 | 144 | 0.89 | 0.05 | 0.68 | 0.05 | 0.68 | 0.05 | 1.13 | 26.3 |
| Conv55 | 64 | 16 | 0.87 | 0.05 | 0.63 | 0.05 | 0.63 | 0.05 | 6.33 | 83.68 |
| Conv56 | 32 | 64 | 0.32 | 0.05 | 0.67 | 0.05 | 0.67 | 0.05 | 20.91 | 46.19 |
| Conv57 | 32 | 288 | 0.64 | 0.05 | 0.83 | 0.05 | 0.83 | 0.05 | 5.09 | 46.19 |
| Conv58 | 128 | 32 | 0.39 | 0.05 | 0.74 | 0.05 | 0.74 | 0.05 | 86.43 | 155.4 |
| Conv59 | 32 | 128 | 0.51 | 0.05 | 0.74 | 0.05 | 0.74 | 0.05 | 4.89 | 46.19 |
| Conv60 | 32 | 288 | 0.46 | 0.05 | 0.83 | 0.05 | 0.83 | 0.05 | 4.81 | 46.19 |
| Conv61 | 128 | 32 | 0.68 | 0.05 | 0.74 | 0.05 | 0.74 | 0.05 | 37.39 | 155.4 |
| Conv62 | 32 | 128 | 0.54 | 0.05 | 0.74 | 0.05 | 0.74 | 0.05 | 5.83 | 46.19 |
| Conv63 | 32 | 288 | 0.62 | 0.05 | 0.83 | 0.05 | 0.83 | 0.05 | 4.73 | 46.19 |
| Conv64 | 128 | 32 | 0.65 | 0.05 | 0.74 | 0.05 | 0.74 | 0.05 | 25.68 | 155.4 |
| Conv65 | 32 | 128 | 0.83 | 0.05 | 0.74 | 0.05 | 0.74 | 0.05 | 3.33 | 46.19 |

| Conv66 | 32 | 288 | 0.84 | 0.05 | 0.83 | 0.05 | 0.83 | 0.05 | 2.21 | 46.19 |
|---|---|---|---|---|---|---|---|---|---|---|
| Conv67 | 128 | 32 | 0.83 | 0.05 | 0.74 | 0.05 | 0.74 | 0.05 | 19.46 | 155.4 |
| Conv68 | 32 | 128 | 0.60 | 0.05 | 0.74 | 0.05 | 0.74 | 0.05 | 3.71 | 46.19 |
| Conv69 | 32 | 288 | 0.61 | 0.05 | 0.83 | 0.05 | 0.83 | 0.05 | 4.04 | 46.19 |
| Conv70 | 128 | 32 | 0.71 | 0.05 | 0.74 | 0.05 | 0.74 | 0.05 | 28.1 | 155.4 |
| Conv71 | 32 | 128 | 0.66 | 0.05 | 0.74 | 0.05 | 0.74 | 0.05 | 2.16 | 46.19 |
| Conv72 | 32 | 288 | 0.66 | 0.05 | 0.83 | 0.05 | 0.83 | 0.05 | 2.64 | 46.19 |
| Conv73 | 128 | 32 | 0.84 | 0.05 | 0.74 | 0.05 | 0.74 | 0.05 | 17.43 | 155.4 |
| Conv74 | 32 | 128 | 0.82 | 0.05 | 0.74 | 0.05 | 0.74 | 0.05 | 1.55 | 46.19 |
| Conv75 | 32 | 288 | 0.84 | 0.05 | 0.83 | 0.05 | 0.83 | 0.05 | 0.91 | 46.19 |
| Conv76 | 128 | 32 | 0.71 | 0.05 | 0.74 | 0.05 | 0.74 | 0.05 | 24.07 | 155.4 |
| Conv77 | 32 | 128 | 0.66 | 0.05 | 0.74 | 0.05 | 0.74 | 0.05 | 5.42 | 46.19 |
| Conv78 | 32 | 288 | 0.70 | 0.05 | 0.83 | 0.05 | 0.83 | 0.05 | 2.84 | 46.19 |
| Conv79 | 128 | 32 | 0.65 | 0.05 | 0.74 | 0.05 | 0.74 | 0.05 | 22.67 | 155.4 |
| Conv80 | 32 | 128 | 0.61 | 0.05 | 0.74 | 0.05 | 0.74 | 0.05 | 9.26 | 46.19 |
| Conv81 | 32 | 288 | 0.61 | 0.05 | 0.83 | 0.05 | 0.83 | 0.05 | 5.61 | 46.19 |
| Conv82 | 128 | 32 | 0.61 | 0.05 | 0.74 | 0.05 | 0.74 | 0.05 | 28.81 | 155.4 |
| Conv83 | 32 | 128 | 0.83 | 0.05 | 0.74 | 0.05 | 0.74 | 0.05 | 2.49 | 46.19 |
| Conv84 | 32 | 288 | 0.81 | 0.05 | 0.83 | 0.05 | 0.83 | 0.05 | 2.07 | 46.19 |
| Conv85 | 128 | 32 | 0.70 | 0.05 | 0.74 | 0.05 | 0.74 | 0.05 | 18.55 | 155.4 |
| Conv86 | 32 | 128 | 0.67 | 0.05 | 0.74 | 0.05 | 0.74 | 0.05 | 4.81 | 46.19 |
| Conv87 | 32 | 288 | 0.62 | 0.05 | 0.83 | 0.05 | 0.83 | 0.05 | 3.47 | 46.19 |
| Conv88 | 128 | 32 | 0.82 | 0.05 | 0.74 | 0.05 | 0.74 | 0.05 | 14.21 | 155.4 |
| Conv89 | 32 | 128 | 0.69 | 0.05 | 0.74 | 0.05 | 0.74 | 0.05 | 4.38 | 46.19 |
| Conv90 | 32 | 288 | 0.84 | 0.05 | 0.83 | 0.05 | 0.83 | 0.05 | 2.32 | 46.19 |
| Conv91 | 128 | 32 | 0.84 | 0.05 | 0.74 | 0.05 | 0.74 | 0.05 | 18.52 | 155.4 |
| Conv92 | 32 | 128 | 0.60 | 0.05 | 0.74 | 0.05 | 0.74 | 0.05 | 4.86 | 46.19 |
| Conv93 | 32 | 288 | 0.84 | 0.05 | 0.83 | 0.05 | 0.83 | 0.05 | 2.01 | 46.19 |
| Conv94 | 128 | 32 | 0.85 | 0.05 | 0.74 | 0.05 | 0.74 | 0.05 | 21.5 | 155.4 |
| Conv95 | 32 | 128 | 0.44 | 0.05 | 0.74 | 0.05 | 0.74 | 0.05 | 7.02 | 46.19 |
| Conv96 | 32 | 288 | 0.61 | 0.05 | 0.83 | 0.05 | 0.83 | 0.05 | 3.26 | 46.19 |
| Conv97 | 128 | 32 | 0.81 | 0.05 | 0.74 | 0.05 | 0.74 | 0.05 | 23.4 | 155.4 |
| Conv98 | 32 | 128 | 0.84 | 0.05 | 0.74 | 0.05 | 0.74 | 0.05 | 2.63 | 46.19 |
| Conv99 | 32 | 288 | 0.69 | 0.05 | 0.83 | 0.05 | 0.83 | 0.05 | 1.53 | 46.19 |
| Conv100 | 128 | 32 | 0.68 | 0.05 | 0.74 | 0.05 | 0.74 | 0.05 | 22.25 | 155.4 |
| Conv101 | 32 | 128 | 0.69 | 0.05 | 0.74 | 0.05 | 0.74 | 0.05 | 3.22 | 46.19 |
| Conv102 | 32 | 288 | 0.85 | 0.05 | 0.83 | 0.05 | 0.83 | 0.05 | 1.5 | 46.19 |
| Conv103 | 128 | 32 | 0.82 | 0.05 | 0.74 | 0.05 | 0.74 | 0.05 | 15.27 | 155.4 |
| Conv104 | 32 | 128 | 0.83 | 0.05 | 0.74 | 0.05 | 0.74 | 0.05 | 3.01 | 46.19 |
| Conv105 | 32 | 288 | 0.85 | 0.05 | 0.83 | 0.05 | 0.83 | 0.05 | 1.79 | 46.19 |
| Conv106 | 128 | 32 | 0.70 | 0.05 | 0.74 | 0.05 | 0.74 | 0.05 | 21.86 | 155.4 |
| Conv107 | 32 | 128 | 0.58 | 0.05 | 0.74 | 0.05 | 0.74 | 0.05 | 7.47 | 46.19 |
| Conv108 | 32 | 288 | 0.81 | 0.05 | 0.83 | 0.05 | 0.83 | 0.05 | 1.14 | 46.19 |
| Conv109 | 128 | 32 | 0.65 | 0.05 | 0.74 | 0.05 | 0.74 | 0.05 | 24.02 | 155.4 |
| Conv110 | 64 | 128 | 0.42 | 0.05 | 0.82 | 0.05 | 0.82 | 0.05 | 10.69 | 83.68 |
| Conv111 | 64 | 576 | 0.66 | 0.05 | 0.97 | 0.05 | 0.97 | 0.05 | 2.19 | 83.68 |
| Conv112 | 256 | 64 | 0.53 | 0.05 | 0.90 | 0.05 | 0.90 | 0.05 | 80.96 | 294.32 |
| Conv113 | 64 | 256 | 0.52 | 0.05 | 0.90 | 0.05 | 0.90 | 0.05 | 1.5 | 83.68 |
| Conv114 | 64 | 576 | 0.83 | 0.05 | 0.97 | 0.05 | 0.97 | 0.05 | 1.43 | 83.68 |
| Conv115 | 256 | 64 | 0.83 | 0.05 | 0.90 | 0.05 | 0.90 | 0.05 | 24.22 | 294.32 |
| Conv116 | 64 | 256 | 0.82 | 0.05 | 0.90 | 0.05 | 0.90 | 0.05 | 1.18 | 83.68 |
| Conv117 | 64 | 576 | 0.84 | 0.05 | 0.97 | 0.05 | 0.97 | 0.05 | 0.89 | 83.68 |
| Conv118 | 256 | 64 | 0.87 | 0.05 | 0.90 | 0.05 | 0.90 | 0.05 | 12.44 | 294.32 |
| Conv119 | 64 | 256 | 0.61 | 0.05 | 0.90 | 0.05 | 0.90 | 0.05 | 1.29 | 83.68 |
| Conv120 | 64 | 576 | 0.62 | 0.05 | 0.97 | 0.05 | 0.97 | 0.05 | 1.29 | 83.68 |
| Conv121 | 256 | 64 | 0.68 | 0.05 | 0.90 | 0.05 | 0.90 | 0.05 | 16.31 | 294.32 |
| Conv122 | 64 | 256 | 0.71 | 0.05 | 0.90 | 0.05 | 0.90 | 0.05 | 1.4 | 83.68 |
| Conv123 | 64 | 576 | 0.87 | 0.05 | 0.97 | 0.05 | 0.97 | 0.05 | 1.46 | 83.68 |
| Conv124 | 256 | 64 | 0.86 | 0.05 | 0.90 | 0.05 | 0.90 | 0.05 | 13.1 | 294.32 |

| Layer | Number | dim | Gaussian | | Mean_Left | | Mean_Right | | Sigma | |
|-------|--------|-----|----------|---------|-----------|---------|------------|---------|-------|---------|
| | | | p-value | c-value | p-value | c-value | p-value | c-value | t-value | c-value |
| Conv125 | 64 | 256 | 0.83 | 0.05 | 0.90 | 0.05 | 0.90 | 0.05 | 1.03 | 83.68 |
| Conv126 | 64 | 576 | 0.89 | 0.05 | 0.97 | 0.05 | 0.97 | 0.05 | 0.67 | 83.68 |
| Conv127 | 256 | 64 | 0.84 | 0.05 | 0.90 | 0.05 | 0.90 | 0.05 | 13.5 | 294.32 |
| Conv128 | 64 | 256 | 0.71 | 0.05 | 0.90 | 0.05 | 0.90 | 0.05 | 1.42 | 83.68 |
| Conv129 | 64 | 576 | 0.86 | 0.05 | 0.97 | 0.05 | 0.97 | 0.05 | 0.45 | 83.68 |
| Conv130 | 256 | 64 | 0.82 | 0.05 | 0.90 | 0.05 | 0.90 | 0.05 | 17.02 | 294.32 |
| Conv131 | 64 | 256 | 0.63 | 0.05 | 0.90 | 0.05 | 0.90 | 0.05 | 1.53 | 83.68 |
| Conv132 | 64 | 576 | 0.88 | 0.05 | 0.97 | 0.05 | 0.97 | 0.05 | 0.62 | 83.68 |
| Conv133 | 256 | 64 | 0.82 | 0.05 | 0.90 | 0.05 | 0.90 | 0.05 | 19.25 | 294.32 |
| Conv134 | 64 | 256 | 0.71 | 0.05 | 0.90 | 0.05 | 0.90 | 0.05 | 1.2 | 83.68 |
| Conv135 | 64 | 576 | 0.88 | 0.05 | 0.97 | 0.05 | 0.97 | 0.05 | 0.83 | 83.68 |
| Conv136 | 256 | 64 | 0.85 | 0.05 | 0.90 | 0.05 | 0.90 | 0.05 | 18.2 | 294.32 |
| Conv137 | 64 | 256 | 0.82 | 0.05 | 0.90 | 0.05 | 0.90 | 0.05 | 0.91 | 83.68 |
| Conv138 | 64 | 576 | 0.88 | 0.05 | 0.97 | 0.05 | 0.97 | 0.05 | 0.59 | 83.68 |
| Conv139 | 256 | 64 | 0.84 | 0.05 | 0.90 | 0.05 | 0.90 | 0.05 | 16.94 | 294.32 |
| Conv140 | 64 | 256 | 0.65 | 0.05 | 0.90 | 0.05 | 0.90 | 0.05 | 1.14 | 83.68 |
| Conv141 | 64 | 576 | 0.86 | 0.05 | 0.97 | 0.05 | 0.97 | 0.05 | 0.87 | 83.68 |
| Conv142 | 256 | 64 | 0.64 | 0.05 | 0.90 | 0.05 | 0.90 | 0.05 | 17.78 | 294.32 |
| Conv143 | 64 | 256 | 0.67 | 0.05 | 0.90 | 0.05 | 0.90 | 0.05 | 1.41 | 83.68 |
| Conv144 | 64 | 576 | 0.87 | 0.05 | 0.97 | 0.05 | 0.97 | 0.05 | 0.78 | 83.68 |
| Conv145 | 256 | 64 | 0.62 | 0.05 | 0.90 | 0.05 | 0.90 | 0.05 | 17.51 | 294.32 |
| Conv146 | 64 | 256 | 0.52 | 0.05 | 0.90 | 0.05 | 0.90 | 0.05 | 1.45 | 83.68 |
| Conv147 | 64 | 576 | 0.86 | 0.05 | 0.97 | 0.05 | 0.97 | 0.05 | 1.19 | 83.68 |
| Conv148 | 256 | 64 | 0.67 | 0.05 | 0.90 | 0.05 | 0.90 | 0.05 | 24.19 | 294.32 |
| Conv149 | 64 | 256 | 0.60 | 0.05 | 0.90 | 0.05 | 0.90 | 0.05 | 0.89 | 83.68 |
| Conv150 | 64 | 576 | 0.83 | 0.05 | 0.97 | 0.05 | 0.97 | 0.05 | 0.97 | 83.68 |
| Conv151 | 256 | 64 | 0.70 | 0.05 | 0.90 | 0.05 | 0.90 | 0.05 | 23.84 | 294.32 |
| Conv152 | 64 | 256 | 0.49 | 0.05 | 0.90 | 0.05 | 0.90 | 0.05 | 1.09 | 83.68 |
| Conv153 | 64 | 576 | 0.83 | 0.05 | 0.97 | 0.05 | 0.97 | 0.05 | 0.88 | 83.68 |
| Conv154 | 256 | 64 | 0.65 | 0.05 | 0.90 | 0.05 | 0.90 | 0.05 | 25.16 | 294.32 |
| Conv155 | 64 | 256 | 0.65 | 0.05 | 0.90 | 0.05 | 0.90 | 0.05 | 1.08 | 83.68 |
| Conv156 | 64 | 576 | 0.83 | 0.05 | 0.97 | 0.05 | 0.97 | 0.05 | 0.74 | 83.68 |
| Conv157 | 256 | 64 | 0.62 | 0.05 | 0.90 | 0.05 | 0.90 | 0.05 | 35.19 | 294.32 |
| Conv158 | 64 | 256 | 0.53 | 0.05 | 0.90 | 0.05 | 0.90 | 0.05 | 1.14 | 83.68 |
| Conv159 | 64 | 576 | 0.83 | 0.05 | 0.97 | 0.05 | 0.97 | 0.05 | 0.64 | 83.68 |
| Conv160 | 256 | 64 | 0.52 | 0.05 | 0.90 | 0.05 | 0.90 | 0.05 | 57.16 | 294.32 |
| Conv161 | 64 | 256 | 0.39 | 0.05 | 0.90 | 0.05 | 0.90 | 0.05 | 1.63 | 83.68 |
| Conv162 | 64 | 576 | 0.86 | 0.05 | 0.97 | 0.05 | 0.97 | 0.05 | 0.53 | 83.68 |
| Conv163 | 256 | 64 | 0.45 | 0.05 | 0.90 | 0.05 | 0.90 | 0.05 | 136.33 | 294.32 |
| **Passing rate** | - | - | 99.39% | | 100.0% | | 100.0% | | 97.55% | |

Table 29: CBAM (ResNet164) Cifar100

| Layer | Number | dim | Gaussian | | Mean_Left | | Mean_Right | | Sigma | |
|-------|--------|-----|----------|---------|-----------|---------|------------|---------|---------|---------|
| | | | p-value | c-value | p-value | c-value | p-value | c-value | t-value | c-value |
| Conv1 | 16 | 27 | 0.02 | 0.05 | 0.58 | 0.05 | 0.58 | 0.05 | 197.16 | 26.3 |
| Conv2 | 16 | 16 | 0.61 | 0.05 | 0.56 | 0.05 | 0.56 | 0.05 | 13.19 | 26.3 |
| Conv3 | 16 | 144 | 0.82 | 0.05 | 0.68 | 0.05 | 0.68 | 0.05 | 2.7 | 26.3 |
| Conv4 | 64 | 16 | 0.71 | 0.05 | 0.63 | 0.05 | 0.63 | 0.05 | 11.87 | 83.68 |
| Conv5 | 16 | 64 | 0.89 | 0.05 | 0.63 | 0.05 | 0.63 | 0.05 | 0.5 | 26.3 |
| Conv6 | 16 | 144 | 0.88 | 0.05 | 0.68 | 0.05 | 0.68 | 0.05 | 0.75 | 26.3 |
| Conv7 | 64 | 16 | 0.88 | 0.05 | 0.63 | 0.05 | 0.63 | 0.05 | 3.67 | 83.68 |
| Conv8 | 16 | 64 | 0.85 | 0.05 | 0.63 | 0.05 | 0.63 | 0.05 | 1.4 | 26.3 |
| Conv9 | 16 | 144 | 0.83 | 0.05 | 0.68 | 0.05 | 0.68 | 0.05 | 1.35 | 26.3 |
| Conv10 | 64 | 16 | 0.68 | 0.05 | 0.63 | 0.05 | 0.63 | 0.05 | 12.85 | 83.68 |

| | | | | | | | | | | |
|---|---|---|---|---|---|---|---|---|---|---|
| Conv11 | 16 | 64 | 0.84 | 0.05 | 0.63 | 0.05 | 0.63 | 0.05 | 1.46 | 26.3 |
| Conv12 | 16 | 144 | 0.82 | 0.05 | 0.68 | 0.05 | 0.68 | 0.05 | 1.31 | 26.3 |
| Conv13 | 64 | 16 | 0.71 | 0.05 | 0.63 | 0.05 | 0.63 | 0.05 | 8.4 | 83.68 |
| Conv14 | 16 | 64 | 0.89 | 0.05 | 0.63 | 0.05 | 0.63 | 0.05 | 0.71 | 26.3 |
| Conv15 | 16 | 144 | 0.91 | 0.05 | 0.68 | 0.05 | 0.68 | 0.05 | 0.28 | 26.3 |
| Conv16 | 64 | 16 | 0.87 | 0.05 | 0.63 | 0.05 | 0.63 | 0.05 | 2.2 | 83.68 |
| Conv17 | 16 | 64 | 0.84 | 0.05 | 0.63 | 0.05 | 0.63 | 0.05 | 1.68 | 26.3 |
| Conv18 | 16 | 144 | 0.86 | 0.05 | 0.68 | 0.05 | 0.68 | 0.05 | 0.97 | 26.3 |
| Conv19 | 64 | 16 | 0.89 | 0.05 | 0.63 | 0.05 | 0.63 | 0.05 | 5.12 | 83.68 |
| Conv20 | 16 | 64 | 0.65 | 0.05 | 0.63 | 0.05 | 0.63 | 0.05 | 2.8 | 26.3 |
| Conv21 | 16 | 144 | 0.54 | 0.05 | 0.68 | 0.05 | 0.68 | 0.05 | 1.76 | 26.3 |
| Conv22 | 64 | 16 | 0.84 | 0.05 | 0.63 | 0.05 | 0.63 | 0.05 | 11.18 | 83.68 |
| Conv23 | 16 | 64 | 0.82 | 0.05 | 0.63 | 0.05 | 0.63 | 0.05 | 2.65 | 26.3 |
| Conv24 | 16 | 144 | 0.65 | 0.05 | 0.68 | 0.05 | 0.68 | 0.05 | 1.47 | 26.3 |
| Conv25 | 64 | 16 | 0.86 | 0.05 | 0.63 | 0.05 | 0.63 | 0.05 | 5.79 | 83.68 |
| Conv26 | 16 | 64 | 0.70 | 0.05 | 0.63 | 0.05 | 0.63 | 0.05 | 3.49 | 26.3 |
| Conv27 | 16 | 144 | 0.86 | 0.05 | 0.68 | 0.05 | 0.68 | 0.05 | 0.96 | 26.3 |
| Conv28 | 64 | 16 | 0.85 | 0.05 | 0.63 | 0.05 | 0.63 | 0.05 | 8.09 | 83.68 |
| Conv29 | 16 | 64 | 0.91 | 0.05 | 0.63 | 0.05 | 0.63 | 0.05 | 1.29 | 26.3 |
| Conv30 | 16 | 144 | 0.91 | 0.05 | 0.68 | 0.05 | 0.68 | 0.05 | 1.01 | 26.3 |
| Conv31 | 64 | 16 | 0.92 | 0.05 | 0.63 | 0.05 | 0.63 | 0.05 | 1.54 | 83.68 |
| Conv32 | 16 | 64 | 0.90 | 0.05 | 0.63 | 0.05 | 0.63 | 0.05 | 0.92 | 26.3 |
| Conv33 | 16 | 144 | 0.91 | 0.05 | 0.68 | 0.05 | 0.68 | 0.05 | 0.4 | 26.3 |
| Conv34 | 64 | 16 | 0.91 | 0.05 | 0.63 | 0.05 | 0.63 | 0.05 | 2.64 | 83.68 |
| Conv35 | 16 | 64 | 0.88 | 0.05 | 0.63 | 0.05 | 0.63 | 0.05 | 1.48 | 26.3 |
| Conv36 | 16 | 144 | 0.92 | 0.05 | 0.68 | 0.05 | 0.68 | 0.05 | 0.17 | 26.3 |
| Conv37 | 64 | 16 | 0.91 | 0.05 | 0.63 | 0.05 | 0.63 | 0.05 | 4.65 | 83.68 |
| Conv38 | 16 | 64 | 0.66 | 0.05 | 0.63 | 0.05 | 0.63 | 0.05 | 4.56 | 26.3 |
| Conv39 | 16 | 144 | 0.67 | 0.05 | 0.68 | 0.05 | 0.68 | 0.05 | 5.55 | 26.3 |
| Conv40 | 64 | 16 | 0.70 | 0.05 | 0.63 | 0.05 | 0.63 | 0.05 | 21.21 | 83.68 |
| Conv41 | 16 | 64 | 0.87 | 0.05 | 0.63 | 0.05 | 0.63 | 0.05 | 0.91 | 26.3 |
| Conv42 | 16 | 144 | 0.93 | 0.05 | 0.68 | 0.05 | 0.68 | 0.05 | 0.13 | 26.3 |
| Conv43 | 64 | 16 | 0.91 | 0.05 | 0.63 | 0.05 | 0.63 | 0.05 | 2.14 | 83.68 |
| Conv44 | 16 | 64 | 0.92 | 0.05 | 0.63 | 0.05 | 0.63 | 0.05 | 0.71 | 26.3 |
| Conv45 | 16 | 144 | 0.93 | 0.05 | 0.68 | 0.05 | 0.68 | 0.05 | 0.33 | 26.3 |
| Conv46 | 64 | 16 | 0.90 | 0.05 | 0.63 | 0.05 | 0.63 | 0.05 | 2.65 | 83.68 |
| Conv47 | 16 | 64 | 0.69 | 0.05 | 0.63 | 0.05 | 0.63 | 0.05 | 3.17 | 26.3 |
| Conv48 | 16 | 144 | 0.86 | 0.05 | 0.68 | 0.05 | 0.68 | 0.05 | 0.91 | 26.3 |
| Conv49 | 64 | 16 | 0.87 | 0.05 | 0.63 | 0.05 | 0.63 | 0.05 | 6.35 | 83.68 |
| Conv50 | 16 | 64 | 0.89 | 0.05 | 0.63 | 0.05 | 0.63 | 0.05 | 1.24 | 26.3 |
| Conv51 | 16 | 144 | 0.89 | 0.05 | 0.68 | 0.05 | 0.68 | 0.05 | 1.15 | 26.3 |
| Conv52 | 64 | 16 | 0.89 | 0.05 | 0.63 | 0.05 | 0.63 | 0.05 | 7.12 | 83.68 |
| Conv53 | 16 | 64 | 0.53 | 0.05 | 0.63 | 0.05 | 0.63 | 0.05 | 5.77 | 26.3 |
| Conv54 | 16 | 144 | 0.56 | 0.05 | 0.68 | 0.05 | 0.68 | 0.05 | 3.64 | 26.3 |
| Conv55 | 64 | 16 | 0.84 | 0.05 | 0.63 | 0.05 | 0.63 | 0.05 | 17.53 | 83.68 |
| Conv56 | 32 | 64 | 0.43 | 0.05 | 0.67 | 0.05 | 0.67 | 0.05 | 18.41 | 46.19 |
| Conv57 | 32 | 288 | 0.54 | 0.05 | 0.83 | 0.05 | 0.83 | 0.05 | 10.34 | 46.19 |
| Conv58 | 128 | 32 | 0.45 | 0.05 | 0.74 | 0.05 | 0.74 | 0.05 | 82.06 | 155.4 |
| Conv59 | 32 | 128 | 0.55 | 0.05 | 0.74 | 0.05 | 0.74 | 0.05 | 6.51 | 46.19 |
| Conv60 | 32 | 288 | 0.64 | 0.05 | 0.83 | 0.05 | 0.83 | 0.05 | 6.29 | 46.19 |
| Conv61 | 128 | 32 | 0.67 | 0.05 | 0.74 | 0.05 | 0.74 | 0.05 | 50.8 | 155.4 |
| Conv62 | 32 | 128 | 0.55 | 0.05 | 0.74 | 0.05 | 0.74 | 0.05 | 4.23 | 46.19 |
| Conv63 | 32 | 288 | 0.48 | 0.05 | 0.83 | 0.05 | 0.83 | 0.05 | 4.17 | 46.19 |
| Conv64 | 128 | 32 | 0.68 | 0.05 | 0.74 | 0.05 | 0.74 | 0.05 | 21.02 | 155.4 |
| Conv65 | 32 | 128 | 0.70 | 0.05 | 0.74 | 0.05 | 0.74 | 0.05 | 1.86 | 46.19 |
| Conv66 | 32 | 288 | 0.84 | 0.05 | 0.83 | 0.05 | 0.83 | 0.05 | 1.58 | 46.19 |
| Conv67 | 128 | 32 | 0.67 | 0.05 | 0.74 | 0.05 | 0.74 | 0.05 | 25.18 | 155.4 |
| Conv68 | 32 | 128 | 0.61 | 0.05 | 0.74 | 0.05 | 0.74 | 0.05 | 6.87 | 46.19 |
| Conv69 | 32 | 288 | 0.70 | 0.05 | 0.83 | 0.05 | 0.83 | 0.05 | 2.08 | 46.19 |

| | | | | | | | | | | |
|---|---|---|---|---|---|---|---|---|---|---|
| Conv70 | 128 | 32 | 0.63 | 0.05 | 0.74 | 0.05 | 0.74 | 0.05 | 36.62 | 155.4 |
| Conv71 | 32 | 128 | 0.62 | 0.05 | 0.74 | 0.05 | 0.74 | 0.05 | 2.77 | 46.19 |
| Conv72 | 32 | 288 | 0.71 | 0.05 | 0.83 | 0.05 | 0.83 | 0.05 | 1.06 | 46.19 |
| Conv73 | 128 | 32 | 0.82 | 0.05 | 0.74 | 0.05 | 0.74 | 0.05 | 12.76 | 155.4 |
| Conv74 | 32 | 128 | 0.61 | 0.05 | 0.74 | 0.05 | 0.74 | 0.05 | 4.29 | 46.19 |
| Conv75 | 32 | 288 | 0.70 | 0.05 | 0.83 | 0.05 | 0.83 | 0.05 | 2.75 | 46.19 |
| Conv76 | 128 | 32 | 0.58 | 0.05 | 0.74 | 0.05 | 0.74 | 0.05 | 36.58 | 155.4 |
| Conv77 | 32 | 128 | 0.81 | 0.05 | 0.74 | 0.05 | 0.74 | 0.05 | 1.8 | 46.19 |
| Conv78 | 32 | 288 | 0.84 | 0.05 | 0.83 | 0.05 | 0.83 | 0.05 | 1.44 | 46.19 |
| Conv79 | 128 | 32 | 0.83 | 0.05 | 0.74 | 0.05 | 0.74 | 0.05 | 22.78 | 155.4 |
| Conv80 | 32 | 128 | 0.88 | 0.05 | 0.74 | 0.05 | 0.74 | 0.05 | 0.69 | 46.19 |
| Conv81 | 32 | 288 | 0.86 | 0.05 | 0.83 | 0.05 | 0.83 | 0.05 | 0.39 | 46.19 |
| Conv82 | 128 | 32 | 0.87 | 0.05 | 0.74 | 0.05 | 0.74 | 0.05 | 12.09 | 155.4 |
| Conv83 | 32 | 128 | 0.85 | 0.05 | 0.74 | 0.05 | 0.74 | 0.05 | 2.09 | 46.19 |
| Conv84 | 32 | 288 | 0.81 | 0.05 | 0.83 | 0.05 | 0.83 | 0.05 | 1.32 | 46.19 |
| Conv85 | 128 | 32 | 0.84 | 0.05 | 0.74 | 0.05 | 0.74 | 0.05 | 11.39 | 155.4 |
| Conv86 | 32 | 128 | 0.57 | 0.05 | 0.74 | 0.05 | 0.74 | 0.05 | 1.48 | 46.19 |
| Conv87 | 32 | 288 | 0.86 | 0.05 | 0.83 | 0.05 | 0.83 | 0.05 | 0.5 | 46.19 |
| Conv88 | 128 | 32 | 0.83 | 0.05 | 0.74 | 0.05 | 0.74 | 0.05 | 12.88 | 155.4 |
| Conv89 | 32 | 128 | 0.57 | 0.05 | 0.74 | 0.05 | 0.74 | 0.05 | 2.43 | 46.19 |
| Conv90 | 32 | 288 | 0.81 | 0.05 | 0.83 | 0.05 | 0.83 | 0.05 | 0.7 | 46.19 |
| Conv91 | 128 | 32 | 0.81 | 0.05 | 0.74 | 0.05 | 0.74 | 0.05 | 12.91 | 155.4 |
| Conv92 | 32 | 128 | 0.82 | 0.05 | 0.74 | 0.05 | 0.74 | 0.05 | 1.18 | 46.19 |
| Conv93 | 32 | 288 | 0.85 | 0.05 | 0.83 | 0.05 | 0.83 | 0.05 | 0.58 | 46.19 |
| Conv94 | 128 | 32 | 0.87 | 0.05 | 0.74 | 0.05 | 0.74 | 0.05 | 13.73 | 155.4 |
| Conv95 | 32 | 128 | 0.86 | 0.05 | 0.74 | 0.05 | 0.74 | 0.05 | 0.97 | 46.19 |
| Conv96 | 32 | 288 | 0.87 | 0.05 | 0.83 | 0.05 | 0.83 | 0.05 | 0.85 | 46.19 |
| Conv97 | 128 | 32 | 0.70 | 0.05 | 0.74 | 0.05 | 0.74 | 0.05 | 14.06 | 155.4 |
| Conv98 | 32 | 128 | 0.85 | 0.05 | 0.74 | 0.05 | 0.74 | 0.05 | 1.07 | 46.19 |
| Conv99 | 32 | 288 | 0.82 | 0.05 | 0.83 | 0.05 | 0.83 | 0.05 | 0.67 | 46.19 |
| Conv100 | 128 | 32 | 0.86 | 0.05 | 0.74 | 0.05 | 0.74 | 0.05 | 10.9 | 155.4 |
| Conv101 | 32 | 128 | 0.65 | 0.05 | 0.74 | 0.05 | 0.74 | 0.05 | 3.05 | 46.19 |
| Conv102 | 32 | 288 | 0.87 | 0.05 | 0.83 | 0.05 | 0.83 | 0.05 | 0.49 | 46.19 |
| Conv103 | 128 | 32 | 0.82 | 0.05 | 0.74 | 0.05 | 0.74 | 0.05 | 14.8 | 155.4 |
| Conv104 | 32 | 128 | 0.87 | 0.05 | 0.74 | 0.05 | 0.74 | 0.05 | 0.42 | 46.19 |
| Conv105 | 32 | 288 | 0.88 | 0.05 | 0.83 | 0.05 | 0.83 | 0.05 | 0.88 | 46.19 |
| Conv106 | 128 | 32 | 0.85 | 0.05 | 0.74 | 0.05 | 0.74 | 0.05 | 10.12 | 155.4 |
| Conv107 | 32 | 128 | 0.81 | 0.05 | 0.74 | 0.05 | 0.74 | 0.05 | 1.94 | 46.19 |
| Conv108 | 32 | 288 | 0.86 | 0.05 | 0.83 | 0.05 | 0.83 | 0.05 | 0.25 | 46.19 |
| Conv109 | 128 | 32 | 0.85 | 0.05 | 0.74 | 0.05 | 0.74 | 0.05 | 17.64 | 155.4 |
| Conv110 | 64 | 128 | 0.57 | 0.05 | 0.82 | 0.05 | 0.82 | 0.05 | 3.04 | 83.68 |
| Conv111 | 64 | 576 | 0.84 | 0.05 | 0.97 | 0.05 | 0.97 | 0.05 | 0.62 | 83.68 |
| Conv112 | 256 | 64 | 0.46 | 0.05 | 0.90 | 0.05 | 0.90 | 0.05 | 46.96 | 294.32 |
| Conv113 | 64 | 256 | 0.70 | 0.05 | 0.90 | 0.05 | 0.90 | 0.05 | 0.81 | 83.68 |
| Conv114 | 64 | 576 | 0.85 | 0.05 | 0.97 | 0.05 | 0.97 | 0.05 | 0.81 | 83.68 |
| Conv115 | 256 | 64 | 0.60 | 0.05 | 0.90 | 0.05 | 0.90 | 0.05 | 13.34 | 294.32 |
| Conv116 | 64 | 256 | 0.84 | 0.05 | 0.90 | 0.05 | 0.90 | 0.05 | 0.78 | 83.68 |
| Conv117 | 64 | 576 | 0.81 | 0.05 | 0.97 | 0.05 | 0.97 | 0.05 | 0.53 | 83.68 |
| Conv118 | 256 | 64 | 0.64 | 0.05 | 0.90 | 0.05 | 0.90 | 0.05 | 12.21 | 294.32 |
| Conv119 | 64 | 256 | 0.65 | 0.05 | 0.90 | 0.05 | 0.90 | 0.05 | 1.11 | 83.68 |
| Conv120 | 64 | 576 | 0.86 | 0.05 | 0.97 | 0.05 | 0.97 | 0.05 | 0.76 | 83.68 |
| Conv121 | 256 | 64 | 0.81 | 0.05 | 0.90 | 0.05 | 0.90 | 0.05 | 10.67 | 294.32 |
| Conv122 | 64 | 256 | 0.69 | 0.05 | 0.90 | 0.05 | 0.90 | 0.05 | 0.65 | 83.68 |
| Conv123 | 64 | 576 | 0.85 | 0.05 | 0.97 | 0.05 | 0.97 | 0.05 | 0.43 | 83.68 |
| Conv124 | 256 | 64 | 0.81 | 0.05 | 0.90 | 0.05 | 0.90 | 0.05 | 10.07 | 294.32 |
| Conv125 | 64 | 256 | 0.81 | 0.05 | 0.90 | 0.05 | 0.90 | 0.05 | 0.76 | 83.68 |
| Conv126 | 64 | 576 | 0.86 | 0.05 | 0.97 | 0.05 | 0.97 | 0.05 | 0.4 | 83.68 |
| Conv127 | 256 | 64 | 0.84 | 0.05 | 0.90 | 0.05 | 0.90 | 0.05 | 10.13 | 294.32 |
| Conv128 | 64 | 256 | 0.84 | 0.05 | 0.90 | 0.05 | 0.90 | 0.05 | 0.64 | 83.68 |

| Conv129 | 64 | 576 | 0.85 | 0.05 | 0.97 | 0.05 | 0.97 | 0.05 | 0.32 | 83.68 |
| Conv130 | 256 | 64 | 0.81 | 0.05 | 0.90 | 0.05 | 0.90 | 0.05 | 11.56 | 294.32 |
| Conv131 | 64 | 256 | 0.81 | 0.05 | 0.90 | 0.05 | 0.90 | 0.05 | 0.75 | 83.68 |
| Conv132 | 64 | 576 | 0.83 | 0.05 | 0.97 | 0.05 | 0.97 | 0.05 | 0.32 | 83.68 |
| Conv133 | 256 | 64 | 0.58 | 0.05 | 0.90 | 0.05 | 0.90 | 0.05 | 13.42 | 294.32 |
| Conv134 | 64 | 256 | 0.69 | 0.05 | 0.90 | 0.05 | 0.90 | 0.05 | 0.67 | 83.68 |
| Conv135 | 64 | 576 | 0.83 | 0.05 | 0.97 | 0.05 | 0.97 | 0.05 | 0.27 | 83.68 |
| Conv136 | 256 | 64 | 0.68 | 0.05 | 0.90 | 0.05 | 0.90 | 0.05 | 11.61 | 294.32 |
| Conv137 | 64 | 256 | 0.65 | 0.05 | 0.90 | 0.05 | 0.90 | 0.05 | 0.65 | 83.68 |
| Conv138 | 64 | 576 | 0.87 | 0.05 | 0.97 | 0.05 | 0.97 | 0.05 | 0.25 | 83.68 |
| Conv139 | 256 | 64 | 0.65 | 0.05 | 0.90 | 0.05 | 0.90 | 0.05 | 12.04 | 294.32 |
| Conv140 | 64 | 256 | 0.82 | 0.05 | 0.90 | 0.05 | 0.90 | 0.05 | 0.43 | 83.68 |
| Conv141 | 64 | 576 | 0.84 | 0.05 | 0.97 | 0.05 | 0.97 | 0.05 | 0.39 | 83.68 |
| Conv142 | 256 | 64 | 0.69 | 0.05 | 0.90 | 0.05 | 0.90 | 0.05 | 13.46 | 294.32 |
| Conv143 | 64 | 256 | 0.81 | 0.05 | 0.90 | 0.05 | 0.90 | 0.05 | 0.51 | 83.68 |
| Conv144 | 64 | 576 | 0.87 | 0.05 | 0.97 | 0.05 | 0.97 | 0.05 | 0.37 | 83.68 |
| Conv145 | 256 | 64 | 0.62 | 0.05 | 0.90 | 0.05 | 0.90 | 0.05 | 14.42 | 294.32 |
| Conv146 | 64 | 256 | 0.67 | 0.05 | 0.90 | 0.05 | 0.90 | 0.05 | 0.54 | 83.68 |
| Conv147 | 64 | 576 | 0.86 | 0.05 | 0.97 | 0.05 | 0.97 | 0.05 | 0.34 | 83.68 |
| Conv148 | 256 | 64 | 0.67 | 0.05 | 0.90 | 0.05 | 0.90 | 0.05 | 15.7 | 294.32 |
| Conv149 | 64 | 256 | 0.60 | 0.05 | 0.90 | 0.05 | 0.90 | 0.05 | 0.52 | 83.68 |
| Conv150 | 64 | 576 | 0.87 | 0.05 | 0.97 | 0.05 | 0.97 | 0.05 | 0.34 | 83.68 |
| Conv151 | 256 | 64 | 0.69 | 0.05 | 0.90 | 0.05 | 0.90 | 0.05 | 18.04 | 294.32 |
| Conv152 | 64 | 256 | 0.54 | 0.05 | 0.90 | 0.05 | 0.90 | 0.05 | 0.74 | 83.68 |
| Conv153 | 64 | 576 | 0.86 | 0.05 | 0.97 | 0.05 | 0.97 | 0.05 | 0.27 | 83.68 |
| Conv154 | 256 | 64 | 0.63 | 0.05 | 0.90 | 0.05 | 0.90 | 0.05 | 22.21 | 294.32 |
| Conv155 | 64 | 256 | 0.63 | 0.05 | 0.90 | 0.05 | 0.90 | 0.05 | 0.82 | 83.68 |
| Conv156 | 64 | 576 | 0.88 | 0.05 | 0.97 | 0.05 | 0.97 | 0.05 | 0.29 | 83.68 |
| Conv157 | 256 | 64 | 0.63 | 0.05 | 0.90 | 0.05 | 0.90 | 0.05 | 23.29 | 294.32 |
| Conv158 | 64 | 256 | 0.65 | 0.05 | 0.90 | 0.05 | 0.90 | 0.05 | 0.76 | 83.68 |
| Conv159 | 64 | 576 | 0.88 | 0.05 | 0.97 | 0.05 | 0.97 | 0.05 | 0.27 | 83.68 |
| Conv160 | 256 | 64 | 0.61 | 0.05 | 0.90 | 0.05 | 0.90 | 0.05 | 25.39 | 294.32 |
| Conv161 | 64 | 256 | 0.63 | 0.05 | 0.90 | 0.05 | 0.90 | 0.05 | 0.84 | 83.68 |
| Conv162 | 64 | 576 | 0.89 | 0.05 | 0.97 | 0.05 | 0.97 | 0.05 | 0.23 | 83.68 |
| Conv163 | 256 | 64 | 0.57 | 0.05 | 0.90 | 0.05 | 0.90 | 0.05 | 37.17 | 294.32 |
| **Passing rate** | - | - | 99.39% | | 100.0% | | 100.0% | | 99.39% | |

Table 30: IEBN (ResNet164) Cifar100

| Layer | Number | dim | Gaussian | | Mean_Left | | Mean_Right | | Sigma | |
|---|---|---|---|---|---|---|---|---|---|---|
| | | | p-value | c-value | p-value | c-value | p-value | c-value | t-value | c-value |
| Conv1 | 16 | 27 | 0.01 | 0.05 | 0.58 | 0.05 | 0.58 | 0.05 | 195.79 | 26.3 |
| Conv2 | 16 | 16 | 0.54 | 0.05 | 0.56 | 0.05 | 0.56 | 0.05 | 4.71 | 26.3 |
| Conv3 | 16 | 144 | 0.68 | 0.05 | 0.68 | 0.05 | 0.68 | 0.05 | 1.02 | 26.3 |
| Conv4 | 64 | 16 | 0.81 | 0.05 | 0.63 | 0.05 | 0.63 | 0.05 | 15.02 | 83.68 |
| Conv5 | 16 | 64 | 0.65 | 0.05 | 0.63 | 0.05 | 0.63 | 0.05 | 0.91 | 26.3 |
| Conv6 | 16 | 144 | 0.85 | 0.05 | 0.68 | 0.05 | 0.68 | 0.05 | 0.77 | 26.3 |
| Conv7 | 64 | 16 | 0.71 | 0.05 | 0.63 | 0.05 | 0.63 | 0.05 | 10.97 | 83.68 |
| Conv8 | 16 | 64 | 0.51 | 0.05 | 0.63 | 0.05 | 0.63 | 0.05 | 2.16 | 26.3 |
| Conv9 | 16 | 144 | 0.67 | 0.05 | 0.68 | 0.05 | 0.68 | 0.05 | 1.29 | 26.3 |
| Conv10 | 64 | 16 | 0.82 | 0.05 | 0.63 | 0.05 | 0.63 | 0.05 | 13.51 | 83.68 |
| Conv11 | 16 | 64 | 0.66 | 0.05 | 0.63 | 0.05 | 0.63 | 0.05 | 1.9 | 26.3 |
| Conv12 | 16 | 144 | 0.80 | 0.05 | 0.68 | 0.05 | 0.68 | 0.05 | 0.85 | 26.3 |
| Conv13 | 64 | 16 | 0.80 | 0.05 | 0.63 | 0.05 | 0.63 | 0.05 | 7.16 | 83.68 |
| Conv14 | 16 | 64 | 0.69 | 0.05 | 0.63 | 0.05 | 0.63 | 0.05 | 2.7 | 26.3 |

| | | | | | | | | | | | |
|---|---|---|---|---|---|---|---|---|---|---|---|
| Conv15 | 16 | 144 | 0.81 | 0.05 | 0.68 | 0.05 | 0.68 | 0.05 | 0.93 | 26.3 |
| Conv16 | 64 | 16 | 0.83 | 0.05 | 0.63 | 0.05 | 0.63 | 0.05 | 14.86 | 83.68 |
| Conv17 | 16 | 64 | 0.69 | 0.05 | 0.63 | 0.05 | 0.63 | 0.05 | 2.31 | 26.3 |
| Conv18 | 16 | 144 | 0.66 | 0.05 | 0.68 | 0.05 | 0.68 | 0.05 | 1.09 | 26.3 |
| Conv19 | 64 | 16 | 0.81 | 0.05 | 0.63 | 0.05 | 0.63 | 0.05 | 11.16 | 83.68 |
| Conv20 | 16 | 64 | 0.82 | 0.05 | 0.63 | 0.05 | 0.63 | 0.05 | 1.1 | 26.3 |
| Conv21 | 16 | 144 | 0.88 | 0.05 | 0.68 | 0.05 | 0.68 | 0.05 | 0.51 | 26.3 |
| Conv22 | 64 | 16 | 0.85 | 0.05 | 0.63 | 0.05 | 0.63 | 0.05 | 8.11 | 83.68 |
| Conv23 | 16 | 64 | 0.86 | 0.05 | 0.63 | 0.05 | 0.63 | 0.05 | 0.49 | 26.3 |
| Conv24 | 16 | 144 | 0.87 | 0.05 | 0.68 | 0.05 | 0.68 | 0.05 | 1.0 | 26.3 |
| Conv25 | 64 | 16 | 0.86 | 0.05 | 0.63 | 0.05 | 0.63 | 0.05 | 10.94 | 83.68 |
| Conv26 | 16 | 64 | 0.71 | 0.05 | 0.63 | 0.05 | 0.63 | 0.05 | 1.04 | 26.3 |
| Conv27 | 16 | 144 | 0.83 | 0.05 | 0.68 | 0.05 | 0.68 | 0.05 | 1.09 | 26.3 |
| Conv28 | 64 | 16 | 0.68 | 0.05 | 0.63 | 0.05 | 0.63 | 0.05 | 15.37 | 83.68 |
| Conv29 | 16 | 64 | 0.83 | 0.05 | 0.63 | 0.05 | 0.63 | 0.05 | 1.2 | 26.3 |
| Conv30 | 16 | 144 | 0.87 | 0.05 | 0.68 | 0.05 | 0.68 | 0.05 | 0.58 | 26.3 |
| Conv31 | 64 | 16 | 0.85 | 0.05 | 0.63 | 0.05 | 0.63 | 0.05 | 9.0 | 83.68 |
| Conv32 | 16 | 64 | 0.70 | 0.05 | 0.63 | 0.05 | 0.63 | 0.05 | 1.73 | 26.3 |
| Conv33 | 16 | 144 | 0.86 | 0.05 | 0.68 | 0.05 | 0.68 | 0.05 | 1.14 | 26.3 |
| Conv34 | 64 | 16 | 0.82 | 0.05 | 0.63 | 0.05 | 0.63 | 0.05 | 9.77 | 83.68 |
| Conv35 | 16 | 64 | 0.66 | 0.05 | 0.63 | 0.05 | 0.63 | 0.05 | 1.8 | 26.3 |
| Conv36 | 16 | 144 | 0.84 | 0.05 | 0.68 | 0.05 | 0.68 | 0.05 | 0.69 | 26.3 |
| Conv37 | 64 | 16 | 0.70 | 0.05 | 0.63 | 0.05 | 0.63 | 0.05 | 8.9 | 83.68 |
| Conv38 | 16 | 64 | 0.59 | 0.05 | 0.63 | 0.05 | 0.63 | 0.05 | 2.08 | 26.3 |
| Conv39 | 16 | 144 | 0.65 | 0.05 | 0.68 | 0.05 | 0.68 | 0.05 | 1.38 | 26.3 |
| Conv40 | 64 | 16 | 0.81 | 0.05 | 0.63 | 0.05 | 0.63 | 0.05 | 10.24 | 83.68 |
| Conv41 | 16 | 64 | 0.63 | 0.05 | 0.63 | 0.05 | 0.63 | 0.05 | 2.02 | 26.3 |
| Conv42 | 16 | 144 | 0.84 | 0.05 | 0.68 | 0.05 | 0.68 | 0.05 | 0.9 | 26.3 |
| Conv43 | 64 | 16 | 0.82 | 0.05 | 0.63 | 0.05 | 0.63 | 0.05 | 17.71 | 83.68 |
| Conv44 | 16 | 64 | 0.81 | 0.05 | 0.63 | 0.05 | 0.63 | 0.05 | 2.86 | 26.3 |
| Conv45 | 16 | 144 | 0.86 | 0.05 | 0.68 | 0.05 | 0.68 | 0.05 | 1.45 | 26.3 |
| Conv46 | 64 | 16 | 0.64 | 0.05 | 0.63 | 0.05 | 0.63 | 0.05 | 19.59 | 83.68 |
| Conv47 | 16 | 64 | 0.83 | 0.05 | 0.63 | 0.05 | 0.63 | 0.05 | 2.27 | 26.3 |
| Conv48 | 16 | 144 | 0.85 | 0.05 | 0.68 | 0.05 | 0.68 | 0.05 | 0.56 | 26.3 |
| Conv49 | 64 | 16 | 0.84 | 0.05 | 0.63 | 0.05 | 0.63 | 0.05 | 8.48 | 83.68 |
| Conv50 | 16 | 64 | 0.63 | 0.05 | 0.63 | 0.05 | 0.63 | 0.05 | 3.11 | 26.3 |
| Conv51 | 16 | 144 | 0.81 | 0.05 | 0.68 | 0.05 | 0.68 | 0.05 | 1.48 | 26.3 |
| Conv52 | 64 | 16 | 0.71 | 0.05 | 0.63 | 0.05 | 0.63 | 0.05 | 17.95 | 83.68 |
| Conv53 | 16 | 64 | 0.64 | 0.05 | 0.63 | 0.05 | 0.63 | 0.05 | 3.24 | 26.3 |
| Conv54 | 16 | 144 | 0.87 | 0.05 | 0.68 | 0.05 | 0.68 | 0.05 | 1.26 | 26.3 |
| Conv55 | 64 | 16 | 0.67 | 0.05 | 0.63 | 0.05 | 0.63 | 0.05 | 12.57 | 83.68 |
| Conv56 | 32 | 64 | 0.49 | 0.05 | 0.67 | 0.05 | 0.67 | 0.05 | 5.98 | 46.19 |
| Conv57 | 32 | 288 | 0.88 | 0.05 | 0.83 | 0.05 | 0.83 | 0.05 | 0.46 | 46.19 |
| Conv58 | 128 | 32 | 0.68 | 0.05 | 0.74 | 0.05 | 0.74 | 0.05 | 22.72 | 155.4 |
| Conv59 | 32 | 128 | 0.88 | 0.05 | 0.74 | 0.05 | 0.74 | 0.05 | 0.28 | 46.19 |
| Conv60 | 32 | 288 | 0.88 | 0.05 | 0.83 | 0.05 | 0.83 | 0.05 | 0.44 | 46.19 |
| Conv61 | 128 | 32 | 0.87 | 0.05 | 0.74 | 0.05 | 0.74 | 0.05 | 5.9 | 155.4 |
| Conv62 | 32 | 128 | 0.86 | 0.05 | 0.74 | 0.05 | 0.74 | 0.05 | 0.89 | 46.19 |
| Conv63 | 32 | 288 | 0.89 | 0.05 | 0.83 | 0.05 | 0.83 | 0.05 | 0.72 | 46.19 |
| Conv64 | 128 | 32 | 0.87 | 0.05 | 0.74 | 0.05 | 0.74 | 0.05 | 5.71 | 155.4 |
| Conv65 | 32 | 128 | 0.90 | 0.05 | 0.74 | 0.05 | 0.74 | 0.05 | 0.35 | 46.19 |
| Conv66 | 32 | 288 | 0.88 | 0.05 | 0.83 | 0.05 | 0.83 | 0.05 | 0.43 | 46.19 |
| Conv67 | 128 | 32 | 0.85 | 0.05 | 0.74 | 0.05 | 0.74 | 0.05 | 4.75 | 155.4 |
| Conv68 | 32 | 128 | 0.85 | 0.05 | 0.74 | 0.05 | 0.74 | 0.05 | 0.67 | 46.19 |
| Conv69 | 32 | 288 | 0.89 | 0.05 | 0.83 | 0.05 | 0.83 | 0.05 | 0.51 | 46.19 |
| Conv70 | 128 | 32 | 0.88 | 0.05 | 0.74 | 0.05 | 0.74 | 0.05 | 5.8 | 155.4 |
| Conv71 | 32 | 128 | 0.84 | 0.05 | 0.74 | 0.05 | 0.74 | 0.05 | 0.97 | 46.19 |
| Conv72 | 32 | 288 | 0.90 | 0.05 | 0.83 | 0.05 | 0.83 | 0.05 | 0.69 | 46.19 |
| Conv73 | 128 | 32 | 0.85 | 0.05 | 0.74 | 0.05 | 0.74 | 0.05 | 5.58 | 155.4 |

| | | | | | | | | | | |
|---|---|---|---|---|---|---|---|---|---|---|
| Conv74 | 32 | 128 | 0.88 | 0.05 | 0.74 | 0.05 | 0.74 | 0.05 | 0.67 | 46.19 |
| Conv75 | 32 | 288 | 0.89 | 0.05 | 0.83 | 0.05 | 0.83 | 0.05 | 0.55 | 46.19 |
| Conv76 | 128 | 32 | 0.89 | 0.05 | 0.74 | 0.05 | 0.74 | 0.05 | 4.63 | 155.4 |
| Conv77 | 32 | 128 | 0.84 | 0.05 | 0.74 | 0.05 | 0.74 | 0.05 | 0.9 | 46.19 |
| Conv78 | 32 | 288 | 0.89 | 0.05 | 0.83 | 0.05 | 0.83 | 0.05 | 0.24 | 46.19 |
| Conv79 | 128 | 32 | 0.87 | 0.05 | 0.74 | 0.05 | 0.74 | 0.05 | 5.47 | 155.4 |
| Conv80 | 32 | 128 | 0.87 | 0.05 | 0.74 | 0.05 | 0.74 | 0.05 | 0.71 | 46.19 |
| Conv81 | 32 | 288 | 0.85 | 0.05 | 0.83 | 0.05 | 0.83 | 0.05 | 0.73 | 46.19 |
| Conv82 | 128 | 32 | 0.88 | 0.05 | 0.74 | 0.05 | 0.74 | 0.05 | 5.61 | 155.4 |
| Conv83 | 32 | 128 | 0.84 | 0.05 | 0.74 | 0.05 | 0.74 | 0.05 | 0.43 | 46.19 |
| Conv84 | 32 | 288 | 0.91 | 0.05 | 0.83 | 0.05 | 0.83 | 0.05 | 0.35 | 46.19 |
| Conv85 | 128 | 32 | 0.87 | 0.05 | 0.74 | 0.05 | 0.74 | 0.05 | 5.8 | 155.4 |
| Conv86 | 32 | 128 | 0.84 | 0.05 | 0.74 | 0.05 | 0.74 | 0.05 | 1.03 | 46.19 |
| Conv87 | 32 | 288 | 0.88 | 0.05 | 0.83 | 0.05 | 0.83 | 0.05 | 0.47 | 46.19 |
| Conv88 | 128 | 32 | 0.88 | 0.05 | 0.74 | 0.05 | 0.74 | 0.05 | 5.64 | 155.4 |
| Conv89 | 32 | 128 | 0.82 | 0.05 | 0.74 | 0.05 | 0.74 | 0.05 | 1.47 | 46.19 |
| Conv90 | 32 | 288 | 0.89 | 0.05 | 0.83 | 0.05 | 0.83 | 0.05 | 0.33 | 46.19 |
| Conv91 | 128 | 32 | 0.84 | 0.05 | 0.74 | 0.05 | 0.74 | 0.05 | 5.74 | 155.4 |
| Conv92 | 32 | 128 | 0.87 | 0.05 | 0.74 | 0.05 | 0.74 | 0.05 | 0.8 | 46.19 |
| Conv93 | 32 | 288 | 0.90 | 0.05 | 0.83 | 0.05 | 0.83 | 0.05 | 0.36 | 46.19 |
| Conv94 | 128 | 32 | 0.88 | 0.05 | 0.74 | 0.05 | 0.74 | 0.05 | 5.16 | 155.4 |
| Conv95 | 32 | 128 | 0.68 | 0.05 | 0.74 | 0.05 | 0.74 | 0.05 | 1.01 | 46.19 |
| Conv96 | 32 | 288 | 0.85 | 0.05 | 0.83 | 0.05 | 0.83 | 0.05 | 0.38 | 46.19 |
| Conv97 | 128 | 32 | 0.82 | 0.05 | 0.74 | 0.05 | 0.74 | 0.05 | 6.46 | 155.4 |
| Conv98 | 32 | 128 | 0.85 | 0.05 | 0.74 | 0.05 | 0.74 | 0.05 | 0.95 | 46.19 |
| Conv99 | 32 | 288 | 0.90 | 0.05 | 0.83 | 0.05 | 0.83 | 0.05 | 0.36 | 46.19 |
| Conv100 | 128 | 32 | 0.84 | 0.05 | 0.74 | 0.05 | 0.74 | 0.05 | 5.54 | 155.4 |
| Conv101 | 32 | 128 | 0.85 | 0.05 | 0.74 | 0.05 | 0.74 | 0.05 | 1.57 | 46.19 |
| Conv102 | 32 | 288 | 0.87 | 0.05 | 0.83 | 0.05 | 0.83 | 0.05 | 0.2 | 46.19 |
| Conv103 | 128 | 32 | 0.87 | 0.05 | 0.74 | 0.05 | 0.74 | 0.05 | 4.57 | 155.4 |
| Conv104 | 32 | 128 | 0.86 | 0.05 | 0.74 | 0.05 | 0.74 | 0.05 | 0.65 | 46.19 |
| Conv105 | 32 | 288 | 0.90 | 0.05 | 0.83 | 0.05 | 0.83 | 0.05 | 0.25 | 46.19 |
| Conv106 | 128 | 32 | 0.86 | 0.05 | 0.74 | 0.05 | 0.74 | 0.05 | 5.92 | 155.4 |
| Conv107 | 32 | 128 | 0.67 | 0.05 | 0.74 | 0.05 | 0.74 | 0.05 | 1.33 | 46.19 |
| Conv108 | 32 | 288 | 0.84 | 0.05 | 0.83 | 0.05 | 0.83 | 0.05 | 0.3 | 46.19 |
| Conv109 | 128 | 32 | 0.87 | 0.05 | 0.74 | 0.05 | 0.74 | 0.05 | 4.94 | 155.4 |
| Conv110 | 64 | 128 | 0.67 | 0.05 | 0.82 | 0.05 | 0.82 | 0.05 | 2.21 | 83.68 |
| Conv111 | 64 | 576 | 0.88 | 0.05 | 0.97 | 0.05 | 0.97 | 0.05 | 0.29 | 83.68 |
| Conv112 | 256 | 64 | 0.64 | 0.05 | 0.90 | 0.05 | 0.90 | 0.05 | 12.03 | 294.32 |
| Conv113 | 64 | 256 | 0.83 | 0.05 | 0.90 | 0.05 | 0.90 | 0.05 | 0.79 | 83.68 |
| Conv114 | 64 | 576 | 0.89 | 0.05 | 0.97 | 0.05 | 0.97 | 0.05 | 0.37 | 83.68 |
| Conv115 | 256 | 64 | 0.83 | 0.05 | 0.90 | 0.05 | 0.90 | 0.05 | 6.38 | 294.32 |
| Conv116 | 64 | 256 | 0.86 | 0.05 | 0.90 | 0.05 | 0.90 | 0.05 | 0.44 | 83.68 |
| Conv117 | 64 | 576 | 0.90 | 0.05 | 0.97 | 0.05 | 0.97 | 0.05 | 0.42 | 83.68 |
| Conv118 | 256 | 64 | 0.81 | 0.05 | 0.90 | 0.05 | 0.90 | 0.05 | 6.49 | 294.32 |
| Conv119 | 64 | 256 | 0.82 | 0.05 | 0.90 | 0.05 | 0.90 | 0.05 | 0.65 | 83.68 |
| Conv120 | 64 | 576 | 0.88 | 0.05 | 0.97 | 0.05 | 0.97 | 0.05 | 0.51 | 83.68 |
| Conv121 | 256 | 64 | 0.87 | 0.05 | 0.90 | 0.05 | 0.90 | 0.05 | 7.01 | 294.32 |
| Conv122 | 64 | 256 | 0.83 | 0.05 | 0.90 | 0.05 | 0.90 | 0.05 | 0.63 | 83.68 |
| Conv123 | 64 | 576 | 0.90 | 0.05 | 0.97 | 0.05 | 0.97 | 0.05 | 0.34 | 83.68 |
| Conv124 | 256 | 64 | 0.85 | 0.05 | 0.90 | 0.05 | 0.90 | 0.05 | 7.76 | 294.32 |
| Conv125 | 64 | 256 | 0.87 | 0.05 | 0.90 | 0.05 | 0.90 | 0.05 | 0.62 | 83.68 |
| Conv126 | 64 | 576 | 0.89 | 0.05 | 0.97 | 0.05 | 0.97 | 0.05 | 0.28 | 83.68 |
| Conv127 | 256 | 64 | 0.84 | 0.05 | 0.90 | 0.05 | 0.90 | 0.05 | 6.24 | 294.32 |
| Conv128 | 64 | 256 | 0.84 | 0.05 | 0.90 | 0.05 | 0.90 | 0.05 | 0.67 | 83.68 |
| Conv129 | 64 | 576 | 0.87 | 0.05 | 0.97 | 0.05 | 0.97 | 0.05 | 0.26 | 83.68 |
| Conv130 | 256 | 64 | 0.84 | 0.05 | 0.90 | 0.05 | 0.90 | 0.05 | 6.71 | 294.32 |
| Conv131 | 64 | 256 | 0.87 | 0.05 | 0.90 | 0.05 | 0.90 | 0.05 | 0.66 | 83.68 |
| Conv132 | 64 | 576 | 0.88 | 0.05 | 0.97 | 0.05 | 0.97 | 0.05 | 0.23 | 83.68 |

| Layer | | | | | | | | | | |
|-------|-----|-----|------|------|------|------|------|------|------|--------|
| Conv133 | 256 | 64 | 0.85 | 0.05 | 0.90 | 0.05 | 0.90 | 0.05 | 7.47 | 294.32 |
| Conv134 | 64 | 256 | 0.85 | 0.05 | 0.90 | 0.05 | 0.90 | 0.05 | 0.47 | 83.68 |
| Conv135 | 64 | 576 | 0.89 | 0.05 | 0.97 | 0.05 | 0.97 | 0.05 | 0.26 | 83.68 |
| Conv136 | 256 | 64 | 0.87 | 0.05 | 0.90 | 0.05 | 0.90 | 0.05 | 6.6 | 294.32 |
| Conv137 | 64 | 256 | 0.87 | 0.05 | 0.90 | 0.05 | 0.90 | 0.05 | 0.54 | 83.68 |
| Conv138 | 64 | 576 | 0.88 | 0.05 | 0.97 | 0.05 | 0.97 | 0.05 | 0.22 | 83.68 |
| Conv139 | 256 | 64 | 0.86 | 0.05 | 0.90 | 0.05 | 0.90 | 0.05 | 7.99 | 294.32 |
| Conv140 | 64 | 256 | 0.83 | 0.05 | 0.90 | 0.05 | 0.90 | 0.05 | 0.63 | 83.68 |
| Conv141 | 64 | 576 | 0.88 | 0.05 | 0.97 | 0.05 | 0.97 | 0.05 | 0.18 | 83.68 |
| Conv142 | 256 | 64 | 0.84 | 0.05 | 0.90 | 0.05 | 0.90 | 0.05 | 8.04 | 294.32 |
| Conv143 | 64 | 256 | 0.83 | 0.05 | 0.90 | 0.05 | 0.90 | 0.05 | 0.5 | 83.68 |
| Conv144 | 64 | 576 | 0.88 | 0.05 | 0.97 | 0.05 | 0.97 | 0.05 | 0.21 | 83.68 |
| Conv145 | 256 | 64 | 0.85 | 0.05 | 0.90 | 0.05 | 0.90 | 0.05 | 7.78 | 294.32 |
| Conv146 | 64 | 256 | 0.83 | 0.05 | 0.90 | 0.05 | 0.90 | 0.05 | 0.43 | 83.68 |
| Conv147 | 64 | 576 | 0.90 | 0.05 | 0.97 | 0.05 | 0.97 | 0.05 | 0.23 | 83.68 |
| Conv148 | 256 | 64 | 0.85 | 0.05 | 0.90 | 0.05 | 0.90 | 0.05 | 7.64 | 294.32 |
| Conv149 | 64 | 256 | 0.83 | 0.05 | 0.90 | 0.05 | 0.90 | 0.05 | 0.49 | 83.68 |
| Conv150 | 64 | 576 | 0.90 | 0.05 | 0.97 | 0.05 | 0.97 | 0.05 | 0.24 | 83.68 |
| Conv151 | 256 | 64 | 0.82 | 0.05 | 0.90 | 0.05 | 0.90 | 0.05 | 8.91 | 294.32 |
| Conv152 | 64 | 256 | 0.83 | 0.05 | 0.90 | 0.05 | 0.90 | 0.05 | 0.75 | 83.68 |
| Conv153 | 64 | 576 | 0.89 | 0.05 | 0.97 | 0.05 | 0.97 | 0.05 | 0.2 | 83.68 |
| Conv154 | 256 | 64 | 0.83 | 0.05 | 0.90 | 0.05 | 0.90 | 0.05 | 8.11 | 294.32 |
| Conv155 | 64 | 256 | 0.83 | 0.05 | 0.90 | 0.05 | 0.90 | 0.05 | 0.5 | 83.68 |
| Conv156 | 64 | 576 | 0.91 | 0.05 | 0.97 | 0.05 | 0.97 | 0.05 | 0.18 | 83.68 |
| Conv157 | 256 | 64 | 0.81 | 0.05 | 0.90 | 0.05 | 0.90 | 0.05 | 8.0 | 294.32 |
| Conv158 | 64 | 256 | 0.83 | 0.05 | 0.90 | 0.05 | 0.90 | 0.05 | 0.63 | 83.68 |
| Conv159 | 64 | 576 | 0.90 | 0.05 | 0.97 | 0.05 | 0.97 | 0.05 | 0.19 | 83.68 |
| Conv160 | 256 | 64 | 0.82 | 0.05 | 0.90 | 0.05 | 0.90 | 0.05 | 7.69 | 294.32 |
| Conv161 | 64 | 256 | 0.81 | 0.05 | 0.90 | 0.05 | 0.90 | 0.05 | 0.87 | 83.68 |
| Conv162 | 64 | 576 | 0.90 | 0.05 | 0.97 | 0.05 | 0.97 | 0.05 | 0.17 | 83.68 |
| Conv163 | 256 | 64 | 0.83 | 0.05 | 0.90 | 0.05 | 0.90 | 0.05 | 9.42 | 294.32 |
| **Passing rate** | - | - | 99.39% | | 100.0% | | 100.0% | | 99.39% | |

Table 31: SGENet (ResNet164) Cifar100

| Layer | Number | dim | Gaussian | | Mean_Left | | Mean_Right | | Sigma | |
|-------|--------|-----|---------|---------|---------|---------|---------|---------|---------|---------|
| | | | p-value | c-value | p-value | c-value | p-value | c-value | t-value | c-value |
| Conv1 | 16 | 27 | 0.00 | 0.05 | 0.58 | 0.05 | 0.58 | 0.05 | 289.86 | 26.3 |
| Conv2 | 16 | 16 | 0.35 | 0.05 | 0.56 | 0.05 | 0.56 | 0.05 | 46.81 | 26.3 |
| Conv3 | 16 | 144 | 0.29 | 0.05 | 0.68 | 0.05 | 0.68 | 0.05 | 25.84 | 26.3 |
| Conv4 | 64 | 16 | 0.47 | 0.05 | 0.63 | 0.05 | 0.63 | 0.05 | 74.29 | 83.68 |
| Conv5 | 16 | 64 | 0.84 | 0.05 | 0.63 | 0.05 | 0.63 | 0.05 | 1.15 | 26.3 |
| Conv6 | 16 | 144 | 0.67 | 0.05 | 0.68 | 0.05 | 0.68 | 0.05 | 1.91 | 26.3 |
| Conv7 | 64 | 16 | 0.48 | 0.05 | 0.63 | 0.05 | 0.63 | 0.05 | 23.94 | 83.68 |
| Conv8 | 16 | 64 | 0.81 | 0.05 | 0.63 | 0.05 | 0.63 | 0.05 | 1.49 | 26.3 |
| Conv9 | 16 | 144 | 0.81 | 0.05 | 0.68 | 0.05 | 0.68 | 0.05 | 2.24 | 26.3 |
| Conv10 | 64 | 16 | 0.71 | 0.05 | 0.63 | 0.05 | 0.63 | 0.05 | 14.43 | 83.68 |
| Conv11 | 16 | 64 | 0.88 | 0.05 | 0.63 | 0.05 | 0.63 | 0.05 | 0.9 | 26.3 |
| Conv12 | 16 | 144 | 0.84 | 0.05 | 0.68 | 0.05 | 0.68 | 0.05 | 0.98 | 26.3 |
| Conv13 | 64 | 16 | 0.84 | 0.05 | 0.63 | 0.05 | 0.63 | 0.05 | 6.21 | 83.68 |
| Conv14 | 16 | 64 | 0.70 | 0.05 | 0.63 | 0.05 | 0.63 | 0.05 | 1.78 | 26.3 |
| Conv15 | 16 | 144 | 0.62 | 0.05 | 0.68 | 0.05 | 0.68 | 0.05 | 2.58 | 26.3 |
| Conv16 | 64 | 16 | 0.68 | 0.05 | 0.63 | 0.05 | 0.63 | 0.05 | 9.12 | 83.68 |
| Conv17 | 16 | 64 | 0.69 | 0.05 | 0.63 | 0.05 | 0.63 | 0.05 | 2.25 | 26.3 |
| Conv18 | 16 | 144 | 0.82 | 0.05 | 0.68 | 0.05 | 0.68 | 0.05 | 0.69 | 26.3 |

| | | | | | | | | | | | |
|---|---|---|---|---|---|---|---|---|---|---|---|
| Conv19 | 64 | 16 | 0.85 | 0.05 | 0.63 | 0.05 | 0.63 | 0.05 | 6.68 | 83.68 |
| Conv20 | 16 | 64 | 0.68 | 0.05 | 0.63 | 0.05 | 0.63 | 0.05 | 3.13 | 26.3 |
| Conv21 | 16 | 144 | 0.81 | 0.05 | 0.68 | 0.05 | 0.68 | 0.05 | 1.92 | 26.3 |
| Conv22 | 64 | 16 | 0.81 | 0.05 | 0.63 | 0.05 | 0.63 | 0.05 | 9.84 | 83.68 |
| Conv23 | 16 | 64 | 0.88 | 0.05 | 0.63 | 0.05 | 0.63 | 0.05 | 0.99 | 26.3 |
| Conv24 | 16 | 144 | 0.91 | 0.05 | 0.68 | 0.05 | 0.68 | 0.05 | 0.26 | 26.3 |
| Conv25 | 64 | 16 | 0.91 | 0.05 | 0.63 | 0.05 | 0.63 | 0.05 | 2.74 | 83.68 |
| Conv26 | 16 | 64 | 0.60 | 0.05 | 0.63 | 0.05 | 0.63 | 0.05 | 3.57 | 26.3 |
| Conv27 | 16 | 144 | 0.85 | 0.05 | 0.68 | 0.05 | 0.68 | 0.05 | 1.47 | 26.3 |
| Conv28 | 64 | 16 | 0.71 | 0.05 | 0.63 | 0.05 | 0.63 | 0.05 | 9.72 | 83.68 |
| Conv29 | 16 | 64 | 0.58 | 0.05 | 0.63 | 0.05 | 0.63 | 0.05 | 3.86 | 26.3 |
| Conv30 | 16 | 144 | 0.52 | 0.05 | 0.68 | 0.05 | 0.68 | 0.05 | 4.55 | 26.3 |
| Conv31 | 64 | 16 | 0.62 | 0.05 | 0.63 | 0.05 | 0.63 | 0.05 | 9.97 | 83.68 |
| Conv32 | 16 | 64 | 0.84 | 0.05 | 0.63 | 0.05 | 0.63 | 0.05 | 1.92 | 26.3 |
| Conv33 | 16 | 144 | 0.86 | 0.05 | 0.68 | 0.05 | 0.68 | 0.05 | 0.87 | 26.3 |
| Conv34 | 64 | 16 | 0.87 | 0.05 | 0.63 | 0.05 | 0.63 | 0.05 | 6.02 | 83.68 |
| Conv35 | 16 | 64 | 0.84 | 0.05 | 0.63 | 0.05 | 0.63 | 0.05 | 1.97 | 26.3 |
| Conv36 | 16 | 144 | 0.83 | 0.05 | 0.68 | 0.05 | 0.68 | 0.05 | 2.01 | 26.3 |
| Conv37 | 64 | 16 | 0.86 | 0.05 | 0.63 | 0.05 | 0.63 | 0.05 | 3.83 | 83.68 |
| Conv38 | 16 | 64 | 0.84 | 0.05 | 0.63 | 0.05 | 0.63 | 0.05 | 1.73 | 26.3 |
| Conv39 | 16 | 144 | 0.83 | 0.05 | 0.68 | 0.05 | 0.68 | 0.05 | 1.99 | 26.3 |
| Conv40 | 64 | 16 | 0.86 | 0.05 | 0.63 | 0.05 | 0.63 | 0.05 | 7.64 | 83.68 |
| Conv41 | 16 | 64 | 0.90 | 0.05 | 0.63 | 0.05 | 0.63 | 0.05 | 1.03 | 26.3 |
| Conv42 | 16 | 144 | 0.91 | 0.05 | 0.68 | 0.05 | 0.68 | 0.05 | 0.3 | 26.3 |
| Conv43 | 64 | 16 | 0.89 | 0.05 | 0.63 | 0.05 | 0.63 | 0.05 | 3.45 | 83.68 |
| Conv44 | 16 | 64 | 0.63 | 0.05 | 0.63 | 0.05 | 0.63 | 0.05 | 3.89 | 26.3 |
| Conv45 | 16 | 144 | 0.61 | 0.05 | 0.68 | 0.05 | 0.68 | 0.05 | 3.72 | 26.3 |
| Conv46 | 64 | 16 | 0.83 | 0.05 | 0.63 | 0.05 | 0.63 | 0.05 | 12.99 | 83.68 |
| Conv47 | 16 | 64 | 0.49 | 0.05 | 0.63 | 0.05 | 0.63 | 0.05 | 4.34 | 26.3 |
| Conv48 | 16 | 144 | 0.53 | 0.05 | 0.68 | 0.05 | 0.68 | 0.05 | 2.15 | 26.3 |
| Conv49 | 64 | 16 | 0.84 | 0.05 | 0.63 | 0.05 | 0.63 | 0.05 | 10.73 | 83.68 |
| Conv50 | 16 | 64 | 0.67 | 0.05 | 0.63 | 0.05 | 0.63 | 0.05 | 2.16 | 26.3 |
| Conv51 | 16 | 144 | 0.83 | 0.05 | 0.68 | 0.05 | 0.68 | 0.05 | 2.72 | 26.3 |
| Conv52 | 64 | 16 | 0.86 | 0.05 | 0.63 | 0.05 | 0.63 | 0.05 | 7.04 | 83.68 |
| Conv53 | 16 | 64 | 0.82 | 0.05 | 0.63 | 0.05 | 0.63 | 0.05 | 4.5 | 26.3 |
| Conv54 | 16 | 144 | 0.69 | 0.05 | 0.68 | 0.05 | 0.68 | 0.05 | 2.71 | 26.3 |
| Conv55 | 64 | 16 | 0.81 | 0.05 | 0.63 | 0.05 | 0.63 | 0.05 | 14.14 | 83.68 |
| Conv56 | 32 | 64 | 0.63 | 0.05 | 0.67 | 0.05 | 0.67 | 0.05 | 11.28 | 46.19 |
| Conv57 | 32 | 288 | 0.65 | 0.05 | 0.83 | 0.05 | 0.83 | 0.05 | 5.96 | 46.19 |
| Conv58 | 128 | 32 | 0.36 | 0.05 | 0.74 | 0.05 | 0.74 | 0.05 | 98.65 | 155.4 |
| Conv59 | 32 | 128 | 0.54 | 0.05 | 0.74 | 0.05 | 0.74 | 0.05 | 5.87 | 46.19 |
| Conv60 | 32 | 288 | 0.52 | 0.05 | 0.83 | 0.05 | 0.83 | 0.05 | 5.99 | 46.19 |
| Conv61 | 128 | 32 | 0.64 | 0.05 | 0.74 | 0.05 | 0.74 | 0.05 | 40.2 | 155.4 |
| Conv62 | 32 | 128 | 0.38 | 0.05 | 0.74 | 0.05 | 0.74 | 0.05 | 3.92 | 46.19 |
| Conv63 | 32 | 288 | 0.53 | 0.05 | 0.83 | 0.05 | 0.83 | 0.05 | 3.51 | 46.19 |
| Conv64 | 128 | 32 | 0.70 | 0.05 | 0.74 | 0.05 | 0.74 | 0.05 | 19.17 | 155.4 |
| Conv65 | 32 | 128 | 0.67 | 0.05 | 0.74 | 0.05 | 0.74 | 0.05 | 1.91 | 46.19 |
| Conv66 | 32 | 288 | 0.67 | 0.05 | 0.83 | 0.05 | 0.83 | 0.05 | 1.78 | 46.19 |
| Conv67 | 128 | 32 | 0.82 | 0.05 | 0.74 | 0.05 | 0.74 | 0.05 | 18.69 | 155.4 |
| Conv68 | 32 | 128 | 0.81 | 0.05 | 0.74 | 0.05 | 0.74 | 0.05 | 2.23 | 46.19 |
| Conv69 | 32 | 288 | 0.66 | 0.05 | 0.83 | 0.05 | 0.83 | 0.05 | 2.37 | 46.19 |
| Conv70 | 128 | 32 | 0.83 | 0.05 | 0.74 | 0.05 | 0.74 | 0.05 | 22.83 | 155.4 |
| Conv71 | 32 | 128 | 0.31 | 0.05 | 0.74 | 0.05 | 0.74 | 0.05 | 7.17 | 46.19 |
| Conv72 | 32 | 288 | 0.59 | 0.05 | 0.83 | 0.05 | 0.83 | 0.05 | 3.55 | 46.19 |
| Conv73 | 128 | 32 | 0.53 | 0.05 | 0.74 | 0.05 | 0.74 | 0.05 | 28.29 | 155.4 |
| Conv74 | 32 | 128 | 0.53 | 0.05 | 0.74 | 0.05 | 0.74 | 0.05 | 5.34 | 46.19 |
| Conv75 | 32 | 288 | 0.64 | 0.05 | 0.83 | 0.05 | 0.83 | 0.05 | 3.14 | 46.19 |
| Conv76 | 128 | 32 | 0.54 | 0.05 | 0.74 | 0.05 | 0.74 | 0.05 | 36.96 | 155.4 |
| Conv77 | 32 | 128 | 0.84 | 0.05 | 0.74 | 0.05 | 0.74 | 0.05 | 2.0 | 46.19 |

| | | | | | | | | | | |
|---|---|---|---|---|---|---|---|---|---|---|
| Conv78 | 32 | 288 | 0.71 | 0.05 | 0.83 | 0.05 | 0.83 | 0.05 | 1.8 | 46.19 |
| Conv79 | 128 | 32 | 0.85 | 0.05 | 0.74 | 0.05 | 0.74 | 0.05 | 15.94 | 155.4 |
| Conv80 | 32 | 128 | 0.67 | 0.05 | 0.74 | 0.05 | 0.74 | 0.05 | 1.48 | 46.19 |
| Conv81 | 32 | 288 | 0.85 | 0.05 | 0.83 | 0.05 | 0.83 | 0.05 | 1.17 | 46.19 |
| Conv82 | 128 | 32 | 0.71 | 0.05 | 0.74 | 0.05 | 0.74 | 0.05 | 13.44 | 155.4 |
| Conv83 | 32 | 128 | 0.84 | 0.05 | 0.74 | 0.05 | 0.74 | 0.05 | 1.66 | 46.19 |
| Conv84 | 32 | 288 | 0.86 | 0.05 | 0.83 | 0.05 | 0.83 | 0.05 | 1.05 | 46.19 |
| Conv85 | 128 | 32 | 0.84 | 0.05 | 0.74 | 0.05 | 0.74 | 0.05 | 10.81 | 155.4 |
| Conv86 | 32 | 128 | 0.70 | 0.05 | 0.74 | 0.05 | 0.74 | 0.05 | 1.56 | 46.19 |
| Conv87 | 32 | 288 | 0.68 | 0.05 | 0.83 | 0.05 | 0.83 | 0.05 | 0.98 | 46.19 |
| Conv88 | 128 | 32 | 0.84 | 0.05 | 0.74 | 0.05 | 0.74 | 0.05 | 12.75 | 155.4 |
| Conv89 | 32 | 128 | 0.82 | 0.05 | 0.74 | 0.05 | 0.74 | 0.05 | 2.06 | 46.19 |
| Conv90 | 32 | 288 | 0.82 | 0.05 | 0.83 | 0.05 | 0.83 | 0.05 | 1.19 | 46.19 |
| Conv91 | 128 | 32 | 0.84 | 0.05 | 0.74 | 0.05 | 0.74 | 0.05 | 10.13 | 155.4 |
| Conv92 | 32 | 128 | 0.83 | 0.05 | 0.74 | 0.05 | 0.74 | 0.05 | 2.47 | 46.19 |
| Conv93 | 32 | 288 | 0.84 | 0.05 | 0.83 | 0.05 | 0.83 | 0.05 | 1.3 | 46.19 |
| Conv94 | 128 | 32 | 0.80 | 0.05 | 0.74 | 0.05 | 0.74 | 0.05 | 17.69 | 155.4 |
| Conv95 | 32 | 128 | 0.87 | 0.05 | 0.74 | 0.05 | 0.74 | 0.05 | 1.15 | 46.19 |
| Conv96 | 32 | 288 | 0.86 | 0.05 | 0.83 | 0.05 | 0.83 | 0.05 | 1.12 | 46.19 |
| Conv97 | 128 | 32 | 0.84 | 0.05 | 0.74 | 0.05 | 0.74 | 0.05 | 11.38 | 155.4 |
| Conv98 | 32 | 128 | 0.83 | 0.05 | 0.74 | 0.05 | 0.74 | 0.05 | 1.54 | 46.19 |
| Conv99 | 32 | 288 | 0.86 | 0.05 | 0.83 | 0.05 | 0.83 | 0.05 | 1.13 | 46.19 |
| Conv100 | 128 | 32 | 0.85 | 0.05 | 0.74 | 0.05 | 0.74 | 0.05 | 16.13 | 155.4 |
| Conv101 | 32 | 128 | 0.59 | 0.05 | 0.74 | 0.05 | 0.74 | 0.05 | 2.81 | 46.19 |
| Conv102 | 32 | 288 | 0.81 | 0.05 | 0.83 | 0.05 | 0.83 | 0.05 | 1.11 | 46.19 |
| Conv103 | 128 | 32 | 0.83 | 0.05 | 0.74 | 0.05 | 0.74 | 0.05 | 11.62 | 155.4 |
| Conv104 | 32 | 128 | 0.71 | 0.05 | 0.74 | 0.05 | 0.74 | 0.05 | 2.22 | 46.19 |
| Conv105 | 32 | 288 | 0.85 | 0.05 | 0.83 | 0.05 | 0.83 | 0.05 | 1.28 | 46.19 |
| Conv106 | 128 | 32 | 0.82 | 0.05 | 0.74 | 0.05 | 0.74 | 0.05 | 16.64 | 155.4 |
| Conv107 | 32 | 128 | 0.68 | 0.05 | 0.74 | 0.05 | 0.74 | 0.05 | 2.62 | 46.19 |
| Conv108 | 32 | 288 | 0.85 | 0.05 | 0.83 | 0.05 | 0.83 | 0.05 | 1.0 | 46.19 |
| Conv109 | 128 | 32 | 0.62 | 0.05 | 0.74 | 0.05 | 0.74 | 0.05 | 18.47 | 155.4 |
| Conv110 | 64 | 128 | 0.58 | 0.05 | 0.82 | 0.05 | 0.82 | 0.05 | 7.8 | 83.68 |
| Conv111 | 64 | 576 | 0.84 | 0.05 | 0.97 | 0.05 | 0.97 | 0.05 | 1.22 | 83.68 |
| Conv112 | 256 | 64 | 0.50 | 0.05 | 0.90 | 0.05 | 0.90 | 0.05 | 71.76 | 294.32 |
| Conv113 | 64 | 256 | 0.84 | 0.05 | 0.90 | 0.05 | 0.90 | 0.05 | 0.75 | 83.68 |
| Conv114 | 64 | 576 | 0.86 | 0.05 | 0.97 | 0.05 | 0.97 | 0.05 | 0.79 | 83.68 |
| Conv115 | 256 | 64 | 0.70 | 0.05 | 0.90 | 0.05 | 0.90 | 0.05 | 15.19 | 294.32 |
| Conv116 | 64 | 256 | 0.82 | 0.05 | 0.90 | 0.05 | 0.90 | 0.05 | 1.21 | 83.68 |
| Conv117 | 64 | 576 | 0.88 | 0.05 | 0.97 | 0.05 | 0.97 | 0.05 | 0.8 | 83.68 |
| Conv118 | 256 | 64 | 0.87 | 0.05 | 0.90 | 0.05 | 0.90 | 0.05 | 15.06 | 294.32 |
| Conv119 | 64 | 256 | 0.62 | 0.05 | 0.90 | 0.05 | 0.90 | 0.05 | 1.12 | 83.68 |
| Conv120 | 64 | 576 | 0.85 | 0.05 | 0.97 | 0.05 | 0.97 | 0.05 | 0.73 | 83.68 |
| Conv121 | 256 | 64 | 0.82 | 0.05 | 0.90 | 0.05 | 0.90 | 0.05 | 13.67 | 294.32 |
| Conv122 | 64 | 256 | 0.59 | 0.05 | 0.90 | 0.05 | 0.90 | 0.05 | 1.17 | 83.68 |
| Conv123 | 64 | 576 | 0.85 | 0.05 | 0.97 | 0.05 | 0.97 | 0.05 | 0.86 | 83.68 |
| Conv124 | 256 | 64 | 0.84 | 0.05 | 0.90 | 0.05 | 0.90 | 0.05 | 15.79 | 294.32 |
| Conv125 | 64 | 256 | 0.84 | 0.05 | 0.90 | 0.05 | 0.90 | 0.05 | 1.12 | 83.68 |
| Conv126 | 64 | 576 | 0.89 | 0.05 | 0.97 | 0.05 | 0.97 | 0.05 | 0.61 | 83.68 |
| Conv127 | 256 | 64 | 0.84 | 0.05 | 0.90 | 0.05 | 0.90 | 0.05 | 10.56 | 294.32 |
| Conv128 | 64 | 256 | 0.85 | 0.05 | 0.90 | 0.05 | 0.90 | 0.05 | 0.83 | 83.68 |
| Conv129 | 64 | 576 | 0.88 | 0.05 | 0.97 | 0.05 | 0.97 | 0.05 | 0.64 | 83.68 |
| Conv130 | 256 | 64 | 0.81 | 0.05 | 0.90 | 0.05 | 0.90 | 0.05 | 13.88 | 294.32 |
| Conv131 | 64 | 256 | 0.85 | 0.05 | 0.90 | 0.05 | 0.90 | 0.05 | 1.07 | 83.68 |
| Conv132 | 64 | 576 | 0.89 | 0.05 | 0.97 | 0.05 | 0.97 | 0.05 | 0.92 | 83.68 |
| Conv133 | 256 | 64 | 0.85 | 0.05 | 0.90 | 0.05 | 0.90 | 0.05 | 12.13 | 294.32 |
| Conv134 | 64 | 256 | 0.84 | 0.05 | 0.90 | 0.05 | 0.90 | 0.05 | 1.46 | 83.68 |
| Conv135 | 64 | 576 | 0.88 | 0.05 | 0.97 | 0.05 | 0.97 | 0.05 | 0.8 | 83.68 |
| Conv136 | 256 | 64 | 0.84 | 0.05 | 0.90 | 0.05 | 0.90 | 0.05 | 13.75 | 294.32 |

| Layer | Number | dim | | | | | | | |
|-------|--------|-----|------|------|------|------|------|------|------|------|
| Conv137 | 64 | 256 | 0.83 | 0.05 | 0.90 | 0.05 | 0.90 | 0.05 | 0.95 | 83.68 |
| Conv138 | 64 | 576 | 0.87 | 0.05 | 0.97 | 0.05 | 0.97 | 0.05 | 0.51 | 83.68 |
| Conv139 | 256 | 64 | 0.84 | 0.05 | 0.90 | 0.05 | 0.90 | 0.05 | 13.21 | 294.32 |
| Conv140 | 64 | 256 | 0.85 | 0.05 | 0.90 | 0.05 | 0.90 | 0.05 | 1.24 | 83.68 |
| Conv141 | 64 | 576 | 0.88 | 0.05 | 0.97 | 0.05 | 0.97 | 0.05 | 0.75 | 83.68 |
| Conv142 | 256 | 64 | 0.81 | 0.05 | 0.90 | 0.05 | 0.90 | 0.05 | 13.98 | 294.32 |
| Conv143 | 64 | 256 | 0.81 | 0.05 | 0.90 | 0.05 | 0.90 | 0.05 | 0.93 | 83.68 |
| Conv144 | 64 | 576 | 0.86 | 0.05 | 0.97 | 0.05 | 0.97 | 0.05 | 0.74 | 83.68 |
| Conv145 | 256 | 64 | 0.83 | 0.05 | 0.90 | 0.05 | 0.90 | 0.05 | 15.25 | 294.32 |
| Conv146 | 64 | 256 | 0.69 | 0.05 | 0.90 | 0.05 | 0.90 | 0.05 | 1.14 | 83.68 |
| Conv147 | 64 | 576 | 0.85 | 0.05 | 0.97 | 0.05 | 0.97 | 0.05 | 0.67 | 83.68 |
| Conv148 | 256 | 64 | 0.82 | 0.05 | 0.90 | 0.05 | 0.90 | 0.05 | 15.25 | 294.32 |
| Conv149 | 64 | 256 | 0.81 | 0.05 | 0.90 | 0.05 | 0.90 | 0.05 | 0.89 | 83.68 |
| Conv150 | 64 | 576 | 0.88 | 0.05 | 0.97 | 0.05 | 0.97 | 0.05 | 0.93 | 83.68 |
| Conv151 | 256 | 64 | 0.71 | 0.05 | 0.90 | 0.05 | 0.90 | 0.05 | 18.2 | 294.32 |
| Conv152 | 64 | 256 | 0.55 | 0.05 | 0.90 | 0.05 | 0.90 | 0.05 | 1.23 | 83.68 |
| Conv153 | 64 | 576 | 0.84 | 0.05 | 0.97 | 0.05 | 0.97 | 0.05 | 0.97 | 83.68 |
| Conv154 | 256 | 64 | 0.66 | 0.05 | 0.90 | 0.05 | 0.90 | 0.05 | 22.91 | 294.32 |
| Conv155 | 64 | 256 | 0.60 | 0.05 | 0.90 | 0.05 | 0.90 | 0.05 | 1.26 | 83.68 |
| Conv156 | 64 | 576 | 0.84 | 0.05 | 0.97 | 0.05 | 0.97 | 0.05 | 0.88 | 83.68 |
| Conv157 | 256 | 64 | 0.62 | 0.05 | 0.90 | 0.05 | 0.90 | 0.05 | 23.61 | 294.32 |
| Conv158 | 64 | 256 | 0.67 | 0.05 | 0.90 | 0.05 | 0.90 | 0.05 | 0.96 | 83.68 |
| Conv159 | 64 | 576 | 0.83 | 0.05 | 0.97 | 0.05 | 0.97 | 0.05 | 0.91 | 83.68 |
| Conv160 | 256 | 64 | 0.61 | 0.05 | 0.90 | 0.05 | 0.90 | 0.05 | 37.47 | 294.32 |
| Conv161 | 64 | 256 | 0.54 | 0.05 | 0.90 | 0.05 | 0.90 | 0.05 | 1.39 | 83.68 |
| Conv162 | 64 | 576 | 0.84 | 0.05 | 0.97 | 0.05 | 0.97 | 0.05 | 0.4 | 83.68 |
| Conv163 | 256 | 64 | 0.37 | 0.05 | 0.90 | 0.05 | 0.90 | 0.05 | 133.74 | 294.32 |
| **Passing rate** | - | - | 99.39% | | 100.0% | | 100.0% | | 98.77% | |

## P.5  INITIALIZATION

config:

`https://github.com/bearpaw/pytorch-classification.`

`https://pytorch.org/docs/master/nn.init.html#nn-init-doc.`

Table 32: Cifar100 kaiming-uniform-ResNet164

| Layer | Number | dim | Gaussian | | Mean_Left | | Mean_Right | | Sigma | |
|-------|--------|-----|---------|---------|---------|---------|---------|---------|---------|---------|
| | | | p-value | c-value | p-value | c-value | p-value | c-value | t-value | c-value |
| Conv1 | 16 | 27 | 0.00 | 0.05 | 0.58 | 0.05 | 0.58 | 0.05 | 425.97 | 26.3 |
| Conv2 | 16 | 16 | 0.12 | 0.05 | 0.56 | 0.05 | 0.56 | 0.05 | 218.1 | 26.3 |
| Conv3 | 16 | 144 | 0.01 | 0.05 | 0.68 | 0.05 | 0.68 | 0.05 | 168.11 | 26.3 |
| Conv4 | 64 | 16 | 0.48 | 0.05 | 0.63 | 0.05 | 0.63 | 0.05 | 64.78 | 83.68 |
| Conv5 | 16 | 64 | 0.20 | 0.05 | 0.63 | 0.05 | 0.63 | 0.05 | 38.28 | 26.3 |
| Conv6 | 16 | 144 | 0.31 | 0.05 | 0.68 | 0.05 | 0.68 | 0.05 | 15.42 | 26.3 |
| Conv7 | 64 | 16 | 0.56 | 0.05 | 0.63 | 0.05 | 0.63 | 0.05 | 41.67 | 83.68 |
| Conv8 | 16 | 64 | 0.29 | 0.05 | 0.63 | 0.05 | 0.63 | 0.05 | 18.91 | 26.3 |
| Conv9 | 16 | 144 | 0.48 | 0.05 | 0.68 | 0.05 | 0.68 | 0.05 | 9.02 | 26.3 |
| Conv10 | 64 | 16 | 0.82 | 0.05 | 0.63 | 0.05 | 0.63 | 0.05 | 9.31 | 83.68 |
| Conv11 | 16 | 64 | 0.42 | 0.05 | 0.63 | 0.05 | 0.63 | 0.05 | 9.62 | 26.3 |
| Conv12 | 16 | 144 | 0.48 | 0.05 | 0.68 | 0.05 | 0.68 | 0.05 | 2.79 | 26.3 |
| Conv13 | 64 | 16 | 0.69 | 0.05 | 0.63 | 0.05 | 0.63 | 0.05 | 13.65 | 83.68 |
| Conv14 | 16 | 64 | 0.59 | 0.05 | 0.63 | 0.05 | 0.63 | 0.05 | 3.04 | 26.3 |
| Conv15 | 16 | 144 | 0.65 | 0.05 | 0.68 | 0.05 | 0.68 | 0.05 | 2.18 | 26.3 |
| Conv16 | 64 | 16 | 0.68 | 0.05 | 0.63 | 0.05 | 0.63 | 0.05 | 11.98 | 83.68 |

| Conv17 | 16 | 64 | 0.67 | 0.05 | 0.63 | 0.05 | 0.63 | 0.05 | 1.2 | 26.3 |
|---|---|---|---|---|---|---|---|---|---|---|
| Conv18 | 16 | 144 | 0.85 | 0.05 | 0.68 | 0.05 | 0.68 | 0.05 | 0.74 | 26.3 |
| Conv19 | 64 | 16 | 0.86 | 0.05 | 0.63 | 0.05 | 0.63 | 0.05 | 7.27 | 83.68 |
| Conv20 | 16 | 64 | 0.58 | 0.05 | 0.63 | 0.05 | 0.63 | 0.05 | 3.37 | 26.3 |
| Conv21 | 16 | 144 | 0.50 | 0.05 | 0.68 | 0.05 | 0.68 | 0.05 | 3.47 | 26.3 |
| Conv22 | 64 | 16 | 0.64 | 0.05 | 0.63 | 0.05 | 0.63 | 0.05 | 19.22 | 83.68 |
| Conv23 | 16 | 64 | 0.60 | 0.05 | 0.63 | 0.05 | 0.63 | 0.05 | 1.89 | 26.3 |
| Conv24 | 16 | 144 | 0.66 | 0.05 | 0.68 | 0.05 | 0.68 | 0.05 | 2.33 | 26.3 |
| Conv25 | 64 | 16 | 0.67 | 0.05 | 0.63 | 0.05 | 0.63 | 0.05 | 10.58 | 83.68 |
| Conv26 | 16 | 64 | 0.56 | 0.05 | 0.63 | 0.05 | 0.63 | 0.05 | 9.27 | 26.3 |
| Conv27 | 16 | 144 | 0.32 | 0.05 | 0.68 | 0.05 | 0.68 | 0.05 | 6.39 | 26.3 |
| Conv28 | 64 | 16 | 0.66 | 0.05 | 0.63 | 0.05 | 0.63 | 0.05 | 15.84 | 83.68 |
| Conv29 | 16 | 64 | 0.81 | 0.05 | 0.63 | 0.05 | 0.63 | 0.05 | 3.21 | 26.3 |
| Conv30 | 16 | 144 | 0.80 | 0.05 | 0.68 | 0.05 | 0.68 | 0.05 | 2.82 | 26.3 |
| Conv31 | 64 | 16 | 0.83 | 0.05 | 0.63 | 0.05 | 0.63 | 0.05 | 12.09 | 83.68 |
| Conv32 | 16 | 64 | 0.64 | 0.05 | 0.63 | 0.05 | 0.63 | 0.05 | 8.7 | 26.3 |
| Conv33 | 16 | 144 | 0.63 | 0.05 | 0.68 | 0.05 | 0.68 | 0.05 | 3.56 | 26.3 |
| Conv34 | 64 | 16 | 0.83 | 0.05 | 0.63 | 0.05 | 0.63 | 0.05 | 12.82 | 83.68 |
| Conv35 | 16 | 64 | 0.38 | 0.05 | 0.63 | 0.05 | 0.63 | 0.05 | 21.3 | 26.3 |
| Conv36 | 16 | 144 | 0.66 | 0.05 | 0.68 | 0.05 | 0.68 | 0.05 | 1.08 | 26.3 |
| Conv37 | 64 | 16 | 0.81 | 0.05 | 0.63 | 0.05 | 0.63 | 0.05 | 13.25 | 83.68 |
| Conv38 | 16 | 64 | 0.63 | 0.05 | 0.63 | 0.05 | 0.63 | 0.05 | 3.26 | 26.3 |
| Conv39 | 16 | 144 | 0.82 | 0.05 | 0.68 | 0.05 | 0.68 | 0.05 | 3.46 | 26.3 |
| Conv40 | 64 | 16 | 0.69 | 0.05 | 0.63 | 0.05 | 0.63 | 0.05 | 14.77 | 83.68 |
| Conv41 | 16 | 64 | 0.71 | 0.05 | 0.63 | 0.05 | 0.63 | 0.05 | 2.69 | 26.3 |
| Conv42 | 16 | 144 | 0.84 | 0.05 | 0.68 | 0.05 | 0.68 | 0.05 | 1.58 | 26.3 |
| Conv43 | 64 | 16 | 0.88 | 0.05 | 0.63 | 0.05 | 0.63 | 0.05 | 10.16 | 83.68 |
| Conv44 | 16 | 64 | 0.57 | 0.05 | 0.63 | 0.05 | 0.63 | 0.05 | 12.25 | 26.3 |
| Conv45 | 16 | 144 | 0.80 | 0.05 | 0.68 | 0.05 | 0.68 | 0.05 | 2.95 | 26.3 |
| Conv46 | 64 | 16 | 0.84 | 0.05 | 0.63 | 0.05 | 0.63 | 0.05 | 12.53 | 83.68 |
| Conv47 | 16 | 64 | 0.81 | 0.05 | 0.63 | 0.05 | 0.63 | 0.05 | 3.93 | 26.3 |
| Conv48 | 16 | 144 | 0.81 | 0.05 | 0.68 | 0.05 | 0.68 | 0.05 | 2.59 | 26.3 |
| Conv49 | 64 | 16 | 0.82 | 0.05 | 0.63 | 0.05 | 0.63 | 0.05 | 8.55 | 83.68 |
| Conv50 | 16 | 64 | 0.67 | 0.05 | 0.63 | 0.05 | 0.63 | 0.05 | 3.59 | 26.3 |
| Conv51 | 16 | 144 | 0.70 | 0.05 | 0.68 | 0.05 | 0.68 | 0.05 | 2.53 | 26.3 |
| Conv52 | 64 | 16 | 0.84 | 0.05 | 0.63 | 0.05 | 0.63 | 0.05 | 14.52 | 83.68 |
| Conv53 | 16 | 64 | 0.63 | 0.05 | 0.63 | 0.05 | 0.63 | 0.05 | 2.31 | 26.3 |
| Conv54 | 16 | 144 | 0.70 | 0.05 | 0.68 | 0.05 | 0.68 | 0.05 | 1.73 | 26.3 |
| Conv55 | 64 | 16 | 0.87 | 0.05 | 0.63 | 0.05 | 0.63 | 0.05 | 10.24 | 83.68 |
| Conv56 | 32 | 64 | 0.55 | 0.05 | 0.67 | 0.05 | 0.67 | 0.05 | 14.4 | 46.19 |
| Conv57 | 32 | 288 | 0.55 | 0.05 | 0.83 | 0.05 | 0.83 | 0.05 | 6.19 | 46.19 |
| Conv58 | 128 | 32 | 0.35 | 0.05 | 0.74 | 0.05 | 0.74 | 0.05 | 73.04 | 155.4 |
| Conv59 | 32 | 128 | 0.61 | 0.05 | 0.74 | 0.05 | 0.74 | 0.05 | 4.02 | 46.19 |
| Conv60 | 32 | 288 | 0.56 | 0.05 | 0.83 | 0.05 | 0.83 | 0.05 | 6.32 | 46.19 |
| Conv61 | 128 | 32 | 0.64 | 0.05 | 0.74 | 0.05 | 0.74 | 0.05 | 34.02 | 155.4 |
| Conv62 | 32 | 128 | 0.53 | 0.05 | 0.74 | 0.05 | 0.74 | 0.05 | 5.78 | 46.19 |
| Conv63 | 32 | 288 | 0.59 | 0.05 | 0.83 | 0.05 | 0.83 | 0.05 | 5.53 | 46.19 |
| Conv64 | 128 | 32 | 0.68 | 0.05 | 0.74 | 0.05 | 0.74 | 0.05 | 25.02 | 155.4 |
| Conv65 | 32 | 128 | 0.83 | 0.05 | 0.74 | 0.05 | 0.74 | 0.05 | 3.52 | 46.19 |
| Conv66 | 32 | 288 | 0.65 | 0.05 | 0.83 | 0.05 | 0.83 | 0.05 | 3.72 | 46.19 |
| Conv67 | 128 | 32 | 0.66 | 0.05 | 0.74 | 0.05 | 0.74 | 0.05 | 17.98 | 155.4 |
| Conv68 | 32 | 128 | 0.69 | 0.05 | 0.74 | 0.05 | 0.74 | 0.05 | 4.34 | 46.19 |
| Conv69 | 32 | 288 | 0.69 | 0.05 | 0.83 | 0.05 | 0.83 | 0.05 | 3.5 | 46.19 |
| Conv70 | 128 | 32 | 0.61 | 0.05 | 0.74 | 0.05 | 0.74 | 0.05 | 27.62 | 155.4 |
| Conv71 | 32 | 128 | 0.81 | 0.05 | 0.74 | 0.05 | 0.74 | 0.05 | 2.84 | 46.19 |
| Conv72 | 32 | 288 | 0.69 | 0.05 | 0.83 | 0.05 | 0.83 | 0.05 | 2.3 | 46.19 |
| Conv73 | 128 | 32 | 0.69 | 0.05 | 0.74 | 0.05 | 0.74 | 0.05 | 20.52 | 155.4 |
| Conv74 | 32 | 128 | 0.81 | 0.05 | 0.74 | 0.05 | 0.74 | 0.05 | 3.83 | 46.19 |
| Conv75 | 32 | 288 | 0.83 | 0.05 | 0.83 | 0.05 | 0.83 | 0.05 | 2.45 | 46.19 |

| | | | | | | | | | | |
|---|---|---|---|---|---|---|---|---|---|---|
| Conv76 | 128 | 32 | 0.84 | 0.05 | 0.74 | 0.05 | 0.74 | 0.05 | 20.47 | 155.4 |
| Conv77 | 32 | 128 | 0.66 | 0.05 | 0.74 | 0.05 | 0.74 | 0.05 | 9.06 | 46.19 |
| Conv78 | 32 | 288 | 0.58 | 0.05 | 0.83 | 0.05 | 0.83 | 0.05 | 4.44 | 46.19 |
| Conv79 | 128 | 32 | 0.60 | 0.05 | 0.74 | 0.05 | 0.74 | 0.05 | 32.04 | 155.4 |
| Conv80 | 32 | 128 | 0.67 | 0.05 | 0.74 | 0.05 | 0.74 | 0.05 | 3.78 | 46.19 |
| Conv81 | 32 | 288 | 0.64 | 0.05 | 0.83 | 0.05 | 0.83 | 0.05 | 2.57 | 46.19 |
| Conv82 | 128 | 32 | 0.65 | 0.05 | 0.74 | 0.05 | 0.74 | 0.05 | 18.89 | 155.4 |
| Conv83 | 32 | 128 | 0.55 | 0.05 | 0.74 | 0.05 | 0.74 | 0.05 | 9.07 | 46.19 |
| Conv84 | 32 | 288 | 0.82 | 0.05 | 0.83 | 0.05 | 0.83 | 0.05 | 1.57 | 46.19 |
| Conv85 | 128 | 32 | 0.70 | 0.05 | 0.74 | 0.05 | 0.74 | 0.05 | 20.75 | 155.4 |
| Conv86 | 32 | 128 | 0.43 | 0.05 | 0.74 | 0.05 | 0.74 | 0.05 | 6.63 | 46.19 |
| Conv87 | 32 | 288 | 0.66 | 0.05 | 0.83 | 0.05 | 0.83 | 0.05 | 3.08 | 46.19 |
| Conv88 | 128 | 32 | 0.70 | 0.05 | 0.74 | 0.05 | 0.74 | 0.05 | 26.59 | 155.4 |
| Conv89 | 32 | 128 | 0.67 | 0.05 | 0.74 | 0.05 | 0.74 | 0.05 | 3.37 | 46.19 |
| Conv90 | 32 | 288 | 0.80 | 0.05 | 0.83 | 0.05 | 0.83 | 0.05 | 2.39 | 46.19 |
| Conv91 | 128 | 32 | 0.68 | 0.05 | 0.74 | 0.05 | 0.74 | 0.05 | 13.65 | 155.4 |
| Conv92 | 32 | 128 | 0.66 | 0.05 | 0.74 | 0.05 | 0.74 | 0.05 | 6.03 | 46.19 |
| Conv93 | 32 | 288 | 0.58 | 0.05 | 0.83 | 0.05 | 0.83 | 0.05 | 2.37 | 46.19 |
| Conv94 | 128 | 32 | 0.81 | 0.05 | 0.74 | 0.05 | 0.74 | 0.05 | 12.2 | 155.4 |
| Conv95 | 32 | 128 | 0.82 | 0.05 | 0.74 | 0.05 | 0.74 | 0.05 | 3.9 | 46.19 |
| Conv96 | 32 | 288 | 0.83 | 0.05 | 0.83 | 0.05 | 0.83 | 0.05 | 1.87 | 46.19 |
| Conv97 | 128 | 32 | 0.80 | 0.05 | 0.74 | 0.05 | 0.74 | 0.05 | 18.32 | 155.4 |
| Conv98 | 32 | 128 | 0.81 | 0.05 | 0.74 | 0.05 | 0.74 | 0.05 | 3.33 | 46.19 |
| Conv99 | 32 | 288 | 0.69 | 0.05 | 0.83 | 0.05 | 0.83 | 0.05 | 2.19 | 46.19 |
| Conv100 | 128 | 32 | 0.81 | 0.05 | 0.74 | 0.05 | 0.74 | 0.05 | 18.19 | 155.4 |
| Conv101 | 32 | 128 | 0.83 | 0.05 | 0.74 | 0.05 | 0.74 | 0.05 | 2.32 | 46.19 |
| Conv102 | 32 | 288 | 0.86 | 0.05 | 0.83 | 0.05 | 0.83 | 0.05 | 1.37 | 46.19 |
| Conv103 | 128 | 32 | 0.69 | 0.05 | 0.74 | 0.05 | 0.74 | 0.05 | 16.8 | 155.4 |
| Conv104 | 32 | 128 | 0.63 | 0.05 | 0.74 | 0.05 | 0.74 | 0.05 | 4.21 | 46.19 |
| Conv105 | 32 | 288 | 0.84 | 0.05 | 0.83 | 0.05 | 0.83 | 0.05 | 2.49 | 46.19 |
| Conv106 | 128 | 32 | 0.62 | 0.05 | 0.74 | 0.05 | 0.74 | 0.05 | 20.93 | 155.4 |
| Conv107 | 32 | 128 | 0.67 | 0.05 | 0.74 | 0.05 | 0.74 | 0.05 | 2.95 | 46.19 |
| Conv108 | 32 | 288 | 0.63 | 0.05 | 0.83 | 0.05 | 0.83 | 0.05 | 1.17 | 46.19 |
| Conv109 | 128 | 32 | 0.65 | 0.05 | 0.74 | 0.05 | 0.74 | 0.05 | 16.5 | 155.4 |
| Conv110 | 64 | 128 | 0.50 | 0.05 | 0.82 | 0.05 | 0.82 | 0.05 | 9.42 | 83.68 |
| Conv111 | 64 | 576 | 0.70 | 0.05 | 0.97 | 0.05 | 0.97 | 0.05 | 2.0 | 83.68 |
| Conv112 | 256 | 64 | 0.57 | 0.05 | 0.90 | 0.05 | 0.90 | 0.05 | 86.43 | 294.32 |
| Conv113 | 64 | 256 | 0.59 | 0.05 | 0.90 | 0.05 | 0.90 | 0.05 | 2.79 | 83.68 |
| Conv114 | 64 | 576 | 0.83 | 0.05 | 0.97 | 0.05 | 0.97 | 0.05 | 1.65 | 83.68 |
| Conv115 | 256 | 64 | 0.68 | 0.05 | 0.90 | 0.05 | 0.90 | 0.05 | 17.79 | 294.32 |
| Conv116 | 64 | 256 | 0.69 | 0.05 | 0.90 | 0.05 | 0.90 | 0.05 | 3.54 | 83.68 |
| Conv117 | 64 | 576 | 0.81 | 0.05 | 0.97 | 0.05 | 0.97 | 0.05 | 2.4 | 83.68 |
| Conv118 | 256 | 64 | 0.70 | 0.05 | 0.90 | 0.05 | 0.90 | 0.05 | 19.61 | 294.32 |
| Conv119 | 64 | 256 | 0.83 | 0.05 | 0.90 | 0.05 | 0.90 | 0.05 | 2.09 | 83.68 |
| Conv120 | 64 | 576 | 0.86 | 0.05 | 0.97 | 0.05 | 0.97 | 0.05 | 1.3 | 83.68 |
| Conv121 | 256 | 64 | 0.82 | 0.05 | 0.90 | 0.05 | 0.90 | 0.05 | 17.89 | 294.32 |
| Conv122 | 64 | 256 | 0.81 | 0.05 | 0.90 | 0.05 | 0.90 | 0.05 | 2.49 | 83.68 |
| Conv123 | 64 | 576 | 0.64 | 0.05 | 0.97 | 0.05 | 0.97 | 0.05 | 1.33 | 83.68 |
| Conv124 | 256 | 64 | 0.70 | 0.05 | 0.90 | 0.05 | 0.90 | 0.05 | 20.03 | 294.32 |
| Conv125 | 64 | 256 | 0.83 | 0.05 | 0.90 | 0.05 | 0.90 | 0.05 | 2.66 | 83.68 |
| Conv126 | 64 | 576 | 0.81 | 0.05 | 0.97 | 0.05 | 0.97 | 0.05 | 1.52 | 83.68 |
| Conv127 | 256 | 64 | 0.63 | 0.05 | 0.90 | 0.05 | 0.90 | 0.05 | 22.08 | 294.32 |
| Conv128 | 64 | 256 | 0.60 | 0.05 | 0.90 | 0.05 | 0.90 | 0.05 | 3.27 | 83.68 |
| Conv129 | 64 | 576 | 0.64 | 0.05 | 0.97 | 0.05 | 0.97 | 0.05 | 2.41 | 83.68 |
| Conv130 | 256 | 64 | 0.65 | 0.05 | 0.90 | 0.05 | 0.90 | 0.05 | 26.37 | 294.32 |
| Conv131 | 64 | 256 | 0.81 | 0.05 | 0.90 | 0.05 | 0.90 | 0.05 | 2.14 | 83.68 |
| Conv132 | 64 | 576 | 0.85 | 0.05 | 0.97 | 0.05 | 0.97 | 0.05 | 1.69 | 83.68 |
| Conv133 | 256 | 64 | 0.71 | 0.05 | 0.90 | 0.05 | 0.90 | 0.05 | 16.51 | 294.32 |
| Conv134 | 64 | 256 | 0.58 | 0.05 | 0.90 | 0.05 | 0.90 | 0.05 | 2.65 | 83.68 |

| Layer | Number | dim | Gaussian | | Mean_Left | | Mean_Right | | Sigma | |
|---|---|---|---|---|---|---|---|---|---|---|
| | | | p-value | c-value | p-value | c-value | p-value | c-value | t-value | c-value |
| Conv135 | 64 | 576 | 0.84 | 0.05 | 0.97 | 0.05 | 0.97 | 0.05 | 2.34 | 83.68 |
| Conv136 | 256 | 64 | 0.83 | 0.05 | 0.90 | 0.05 | 0.90 | 0.05 | 21.06 | 294.32 |
| Conv137 | 64 | 256 | 0.81 | 0.05 | 0.90 | 0.05 | 0.90 | 0.05 | 2.17 | 83.68 |
| Conv138 | 64 | 576 | 0.82 | 0.05 | 0.97 | 0.05 | 0.97 | 0.05 | 1.3 | 83.68 |
| Conv139 | 256 | 64 | 0.66 | 0.05 | 0.90 | 0.05 | 0.90 | 0.05 | 23.37 | 294.32 |
| Conv140 | 64 | 256 | 0.81 | 0.05 | 0.90 | 0.05 | 0.90 | 0.05 | 1.45 | 83.68 |
| Conv141 | 64 | 576 | 0.85 | 0.05 | 0.97 | 0.05 | 0.97 | 0.05 | 1.02 | 83.68 |
| Conv142 | 256 | 64 | 0.70 | 0.05 | 0.90 | 0.05 | 0.90 | 0.05 | 24.32 | 294.32 |
| Conv143 | 64 | 256 | 0.68 | 0.05 | 0.90 | 0.05 | 0.90 | 0.05 | 2.39 | 83.68 |
| Conv144 | 64 | 576 | 0.83 | 0.05 | 0.97 | 0.05 | 0.97 | 0.05 | 1.32 | 83.68 |
| Conv145 | 256 | 64 | 0.81 | 0.05 | 0.90 | 0.05 | 0.90 | 0.05 | 26.7 | 294.32 |
| Conv146 | 64 | 256 | 0.68 | 0.05 | 0.90 | 0.05 | 0.90 | 0.05 | 2.43 | 83.68 |
| Conv147 | 64 | 576 | 0.82 | 0.05 | 0.97 | 0.05 | 0.97 | 0.05 | 1.31 | 83.68 |
| Conv148 | 256 | 64 | 0.65 | 0.05 | 0.90 | 0.05 | 0.90 | 0.05 | 29.8 | 294.32 |
| Conv149 | 64 | 256 | 0.62 | 0.05 | 0.90 | 0.05 | 0.90 | 0.05 | 2.1 | 83.68 |
| Conv150 | 64 | 576 | 0.83 | 0.05 | 0.97 | 0.05 | 0.97 | 0.05 | 1.14 | 83.68 |
| Conv151 | 256 | 64 | 0.62 | 0.05 | 0.90 | 0.05 | 0.90 | 0.05 | 35.97 | 294.32 |
| Conv152 | 64 | 256 | 0.64 | 0.05 | 0.90 | 0.05 | 0.90 | 0.05 | 1.69 | 83.68 |
| Conv153 | 64 | 576 | 0.84 | 0.05 | 0.97 | 0.05 | 0.97 | 0.05 | 1.07 | 83.68 |
| Conv154 | 256 | 64 | 0.64 | 0.05 | 0.90 | 0.05 | 0.90 | 0.05 | 27.5 | 294.32 |
| Conv155 | 64 | 256 | 0.59 | 0.05 | 0.90 | 0.05 | 0.90 | 0.05 | 2.52 | 83.68 |
| Conv156 | 64 | 576 | 0.81 | 0.05 | 0.97 | 0.05 | 0.97 | 0.05 | 1.11 | 83.68 |
| Conv157 | 256 | 64 | 0.59 | 0.05 | 0.90 | 0.05 | 0.90 | 0.05 | 32.33 | 294.32 |
| Conv158 | 64 | 256 | 0.60 | 0.05 | 0.90 | 0.05 | 0.90 | 0.05 | 1.95 | 83.68 |
| Conv159 | 64 | 576 | 0.84 | 0.05 | 0.97 | 0.05 | 0.97 | 0.05 | 1.16 | 83.68 |
| Conv160 | 256 | 64 | 0.56 | 0.05 | 0.90 | 0.05 | 0.90 | 0.05 | 50.72 | 294.32 |
| Conv161 | 64 | 256 | 0.41 | 0.05 | 0.90 | 0.05 | 0.90 | 0.05 | 2.61 | 83.68 |
| Conv162 | 64 | 576 | 0.82 | 0.05 | 0.97 | 0.05 | 0.97 | 0.05 | 0.61 | 83.68 |
| Conv163 | 256 | 64 | 0.33 | 0.05 | 0.90 | 0.05 | 0.90 | 0.05 | 194.0 | 294.32 |
| **Passing rate** | - | - | 98.77% | | 100.0% | | 100.0% | | 97.55% | |

Table 33: Cifar100 Xavier-normal-ResNet164

| Layer | Number | dim | Gaussian | | Mean_Left | | Mean_Right | | Sigma | |
|---|---|---|---|---|---|---|---|---|---|---|
| | | | p-value | c-value | p-value | c-value | p-value | c-value | t-value | c-value |
| Conv1 | 16 | 27 | 0.00 | 0.05 | 0.58 | 0.05 | 0.42 | 0.05 | 4100.8 | 26.3 |
| Conv2 | 16 | 16 | 0.07 | 0.05 | 0.56 | 0.05 | 0.56 | 0.05 | 263.78 | 26.3 |
| Conv3 | 16 | 144 | 0.03 | 0.05 | 0.68 | 0.05 | 0.68 | 0.05 | 124.29 | 26.3 |
| Conv4 | 64 | 16 | 0.24 | 0.05 | 0.63 | 0.05 | 0.63 | 0.05 | 218.9 | 83.68 |
| Conv5 | 16 | 64 | 0.28 | 0.05 | 0.63 | 0.05 | 0.63 | 0.05 | 42.91 | 26.3 |
| Conv6 | 16 | 144 | 0.25 | 0.05 | 0.68 | 0.05 | 0.68 | 0.05 | 29.6 | 26.3 |
| Conv7 | 64 | 16 | 0.41 | 0.05 | 0.63 | 0.05 | 0.63 | 0.05 | 63.85 | 83.68 |
| Conv8 | 16 | 64 | 0.38 | 0.05 | 0.63 | 0.05 | 0.63 | 0.05 | 10.79 | 26.3 |
| Conv9 | 16 | 144 | 0.34 | 0.05 | 0.68 | 0.05 | 0.68 | 0.05 | 15.78 | 26.3 |
| Conv10 | 64 | 16 | 0.59 | 0.05 | 0.63 | 0.05 | 0.63 | 0.05 | 40.51 | 83.68 |
| Conv11 | 16 | 64 | 0.41 | 0.05 | 0.63 | 0.05 | 0.63 | 0.05 | 9.5 | 26.3 |
| Conv12 | 16 | 144 | 0.43 | 0.05 | 0.68 | 0.05 | 0.68 | 0.05 | 8.63 | 26.3 |
| Conv13 | 64 | 16 | 0.55 | 0.05 | 0.63 | 0.05 | 0.63 | 0.05 | 29.22 | 83.68 |
| Conv14 | 16 | 64 | 0.47 | 0.05 | 0.63 | 0.05 | 0.63 | 0.05 | 3.81 | 26.3 |
| Conv15 | 16 | 144 | 0.57 | 0.05 | 0.68 | 0.05 | 0.68 | 0.05 | 1.54 | 26.3 |
| Conv16 | 64 | 16 | 0.80 | 0.05 | 0.63 | 0.05 | 0.63 | 0.05 | 13.88 | 83.68 |
| Conv17 | 16 | 64 | 0.68 | 0.05 | 0.63 | 0.05 | 0.63 | 0.05 | 1.34 | 26.3 |
| Conv18 | 16 | 144 | 0.83 | 0.05 | 0.68 | 0.05 | 0.68 | 0.05 | 1.23 | 26.3 |
| Conv19 | 64 | 16 | 0.71 | 0.05 | 0.63 | 0.05 | 0.63 | 0.05 | 7.88 | 83.68 |
| Conv20 | 16 | 64 | 0.70 | 0.05 | 0.63 | 0.05 | 0.63 | 0.05 | 0.98 | 26.3 |

| | | | | | | | | | | |
|---|---|---|---|---|---|---|---|---|---|---|
| Conv21 | 16 | 144 | 0.81 | 0.05 | 0.68 | 0.05 | 0.68 | 0.05 | 0.78 | 26.3 |
| Conv22 | 64 | 16 | 0.83 | 0.05 | 0.63 | 0.05 | 0.63 | 0.05 | 7.01 | 83.68 |
| Conv23 | 16 | 64 | 0.48 | 0.05 | 0.63 | 0.05 | 0.63 | 0.05 | 4.48 | 26.3 |
| Conv24 | 16 | 144 | 0.53 | 0.05 | 0.68 | 0.05 | 0.68 | 0.05 | 2.56 | 26.3 |
| Conv25 | 64 | 16 | 0.48 | 0.05 | 0.63 | 0.05 | 0.63 | 0.05 | 29.86 | 83.68 |
| Conv26 | 16 | 64 | 0.84 | 0.05 | 0.63 | 0.05 | 0.63 | 0.05 | 0.63 | 26.3 |
| Conv27 | 16 | 144 | 0.87 | 0.05 | 0.68 | 0.05 | 0.68 | 0.05 | 0.48 | 26.3 |
| Conv28 | 64 | 16 | 0.92 | 0.05 | 0.63 | 0.05 | 0.63 | 0.05 | 2.04 | 83.68 |
| Conv29 | 16 | 64 | 0.62 | 0.05 | 0.63 | 0.05 | 0.63 | 0.05 | 5.32 | 26.3 |
| Conv30 | 16 | 144 | 0.53 | 0.05 | 0.68 | 0.05 | 0.68 | 0.05 | 3.17 | 26.3 |
| Conv31 | 64 | 16 | 0.56 | 0.05 | 0.63 | 0.05 | 0.63 | 0.05 | 23.86 | 83.68 |
| Conv32 | 16 | 64 | 0.61 | 0.05 | 0.63 | 0.05 | 0.63 | 0.05 | 4.21 | 26.3 |
| Conv33 | 16 | 144 | 0.53 | 0.05 | 0.68 | 0.05 | 0.68 | 0.05 | 4.85 | 26.3 |
| Conv34 | 64 | 16 | 0.65 | 0.05 | 0.63 | 0.05 | 0.63 | 0.05 | 12.41 | 83.68 |
| Conv35 | 16 | 64 | 0.68 | 0.05 | 0.63 | 0.05 | 0.63 | 0.05 | 1.76 | 26.3 |
| Conv36 | 16 | 144 | 0.89 | 0.05 | 0.68 | 0.05 | 0.68 | 0.05 | 0.46 | 26.3 |
| Conv37 | 64 | 16 | 0.85 | 0.05 | 0.63 | 0.05 | 0.63 | 0.05 | 4.31 | 83.68 |
| Conv38 | 16 | 64 | 0.40 | 0.05 | 0.63 | 0.05 | 0.63 | 0.05 | 10.81 | 26.3 |
| Conv39 | 16 | 144 | 0.43 | 0.05 | 0.68 | 0.05 | 0.68 | 0.05 | 11.21 | 26.3 |
| Conv40 | 64 | 16 | 0.66 | 0.05 | 0.63 | 0.05 | 0.63 | 0.05 | 21.52 | 83.68 |
| Conv41 | 16 | 64 | 0.85 | 0.05 | 0.63 | 0.05 | 0.63 | 0.05 | 1.27 | 26.3 |
| Conv42 | 16 | 144 | 0.89 | 0.05 | 0.68 | 0.05 | 0.68 | 0.05 | 0.7 | 26.3 |
| Conv43 | 64 | 16 | 0.86 | 0.05 | 0.63 | 0.05 | 0.63 | 0.05 | 4.92 | 83.68 |
| Conv44 | 16 | 64 | 0.80 | 0.05 | 0.63 | 0.05 | 0.63 | 0.05 | 3.44 | 26.3 |
| Conv45 | 16 | 144 | 0.65 | 0.05 | 0.68 | 0.05 | 0.68 | 0.05 | 3.32 | 26.3 |
| Conv46 | 64 | 16 | 0.67 | 0.05 | 0.63 | 0.05 | 0.63 | 0.05 | 20.05 | 83.68 |
| Conv47 | 16 | 64 | 0.62 | 0.05 | 0.63 | 0.05 | 0.63 | 0.05 | 5.66 | 26.3 |
| Conv48 | 16 | 144 | 0.67 | 0.05 | 0.68 | 0.05 | 0.68 | 0.05 | 4.6 | 26.3 |
| Conv49 | 64 | 16 | 0.81 | 0.05 | 0.63 | 0.05 | 0.63 | 0.05 | 13.5 | 83.68 |
| Conv50 | 16 | 64 | 0.55 | 0.05 | 0.63 | 0.05 | 0.63 | 0.05 | 6.88 | 26.3 |
| Conv51 | 16 | 144 | 0.56 | 0.05 | 0.68 | 0.05 | 0.68 | 0.05 | 6.35 | 26.3 |
| Conv52 | 64 | 16 | 0.59 | 0.05 | 0.63 | 0.05 | 0.63 | 0.05 | 26.53 | 83.68 |
| Conv53 | 16 | 64 | 0.66 | 0.05 | 0.63 | 0.05 | 0.63 | 0.05 | 11.71 | 26.3 |
| Conv54 | 16 | 144 | 0.56 | 0.05 | 0.68 | 0.05 | 0.68 | 0.05 | 5.5 | 26.3 |
| Conv55 | 64 | 16 | 0.55 | 0.05 | 0.63 | 0.05 | 0.63 | 0.05 | 46.57 | 83.68 |
| Conv56 | 32 | 64 | 0.33 | 0.05 | 0.67 | 0.05 | 0.67 | 0.05 | 32.25 | 46.19 |
| Conv57 | 32 | 288 | 0.55 | 0.05 | 0.83 | 0.05 | 0.83 | 0.05 | 3.62 | 46.19 |
| Conv58 | 128 | 32 | 0.42 | 0.05 | 0.74 | 0.05 | 0.74 | 0.05 | 78.52 | 155.4 |
| Conv59 | 32 | 128 | 0.66 | 0.05 | 0.74 | 0.05 | 0.74 | 0.05 | 2.9 | 46.19 |
| Conv60 | 32 | 288 | 0.53 | 0.05 | 0.83 | 0.05 | 0.83 | 0.05 | 4.46 | 46.19 |
| Conv61 | 128 | 32 | 0.60 | 0.05 | 0.74 | 0.05 | 0.74 | 0.05 | 26.11 | 155.4 |
| Conv62 | 32 | 128 | 0.57 | 0.05 | 0.74 | 0.05 | 0.74 | 0.05 | 6.41 | 46.19 |
| Conv63 | 32 | 288 | 0.53 | 0.05 | 0.83 | 0.05 | 0.83 | 0.05 | 5.29 | 46.19 |
| Conv64 | 128 | 32 | 0.67 | 0.05 | 0.74 | 0.05 | 0.74 | 0.05 | 40.74 | 155.4 |
| Conv65 | 32 | 128 | 0.60 | 0.05 | 0.74 | 0.05 | 0.74 | 0.05 | 5.03 | 46.19 |
| Conv66 | 32 | 288 | 0.60 | 0.05 | 0.83 | 0.05 | 0.83 | 0.05 | 4.94 | 46.19 |
| Conv67 | 128 | 32 | 0.68 | 0.05 | 0.74 | 0.05 | 0.74 | 0.05 | 27.32 | 155.4 |
| Conv68 | 32 | 128 | 0.85 | 0.05 | 0.74 | 0.05 | 0.74 | 0.05 | 2.84 | 46.19 |
| Conv69 | 32 | 288 | 0.81 | 0.05 | 0.83 | 0.05 | 0.83 | 0.05 | 2.45 | 46.19 |
| Conv70 | 128 | 32 | 0.82 | 0.05 | 0.74 | 0.05 | 0.74 | 0.05 | 17.34 | 155.4 |
| Conv71 | 32 | 128 | 0.83 | 0.05 | 0.74 | 0.05 | 0.74 | 0.05 | 3.08 | 46.19 |
| Conv72 | 32 | 288 | 0.66 | 0.05 | 0.83 | 0.05 | 0.83 | 0.05 | 2.93 | 46.19 |
| Conv73 | 128 | 32 | 0.83 | 0.05 | 0.74 | 0.05 | 0.74 | 0.05 | 15.7 | 155.4 |
| Conv74 | 32 | 128 | 0.63 | 0.05 | 0.74 | 0.05 | 0.74 | 0.05 | 3.95 | 46.19 |
| Conv75 | 32 | 288 | 0.67 | 0.05 | 0.83 | 0.05 | 0.83 | 0.05 | 2.9 | 46.19 |
| Conv76 | 128 | 32 | 0.63 | 0.05 | 0.74 | 0.05 | 0.74 | 0.05 | 21.4 | 155.4 |
| Conv77 | 32 | 128 | 0.83 | 0.05 | 0.74 | 0.05 | 0.74 | 0.05 | 1.55 | 46.19 |
| Conv78 | 32 | 288 | 0.63 | 0.05 | 0.83 | 0.05 | 0.83 | 0.05 | 1.41 | 46.19 |
| Conv79 | 128 | 32 | 0.80 | 0.05 | 0.74 | 0.05 | 0.74 | 0.05 | 16.74 | 155.4 |

| | | | | | | | | | | |
|---|---|---|---|---|---|---|---|---|---|---|
| Conv80 | 32 | 128 | 0.84 | 0.05 | 0.74 | 0.05 | 0.74 | 0.05 | 1.6 | 46.19 |
| Conv81 | 32 | 288 | 0.84 | 0.05 | 0.83 | 0.05 | 0.83 | 0.05 | 2.41 | 46.19 |
| Conv82 | 128 | 32 | 0.84 | 0.05 | 0.74 | 0.05 | 0.74 | 0.05 | 12.17 | 155.4 |
| Conv83 | 32 | 128 | 0.80 | 0.05 | 0.74 | 0.05 | 0.74 | 0.05 | 3.14 | 46.19 |
| Conv84 | 32 | 288 | 0.83 | 0.05 | 0.83 | 0.05 | 0.83 | 0.05 | 1.82 | 46.19 |
| Conv85 | 128 | 32 | 0.85 | 0.05 | 0.74 | 0.05 | 0.74 | 0.05 | 16.09 | 155.4 |
| Conv86 | 32 | 128 | 0.46 | 0.05 | 0.74 | 0.05 | 0.74 | 0.05 | 7.45 | 46.19 |
| Conv87 | 32 | 288 | 0.61 | 0.05 | 0.83 | 0.05 | 0.83 | 0.05 | 4.67 | 46.19 |
| Conv88 | 128 | 32 | 0.82 | 0.05 | 0.74 | 0.05 | 0.74 | 0.05 | 26.75 | 155.4 |
| Conv89 | 32 | 128 | 0.61 | 0.05 | 0.74 | 0.05 | 0.74 | 0.05 | 5.19 | 46.19 |
| Conv90 | 32 | 288 | 0.68 | 0.05 | 0.83 | 0.05 | 0.83 | 0.05 | 3.7 | 46.19 |
| Conv91 | 128 | 32 | 0.67 | 0.05 | 0.74 | 0.05 | 0.74 | 0.05 | 32.92 | 155.4 |
| Conv92 | 32 | 128 | 0.86 | 0.05 | 0.74 | 0.05 | 0.74 | 0.05 | 2.79 | 46.19 |
| Conv93 | 32 | 288 | 0.83 | 0.05 | 0.83 | 0.05 | 0.83 | 0.05 | 1.96 | 46.19 |
| Conv94 | 128 | 32 | 0.81 | 0.05 | 0.74 | 0.05 | 0.74 | 0.05 | 16.14 | 155.4 |
| Conv95 | 32 | 128 | 0.42 | 0.05 | 0.74 | 0.05 | 0.74 | 0.05 | 4.51 | 46.19 |
| Conv96 | 32 | 288 | 0.68 | 0.05 | 0.83 | 0.05 | 0.83 | 0.05 | 2.83 | 46.19 |
| Conv97 | 128 | 32 | 0.61 | 0.05 | 0.74 | 0.05 | 0.74 | 0.05 | 22.18 | 155.4 |
| Conv98 | 32 | 128 | 0.80 | 0.05 | 0.74 | 0.05 | 0.74 | 0.05 | 1.75 | 46.19 |
| Conv99 | 32 | 288 | 0.86 | 0.05 | 0.83 | 0.05 | 0.83 | 0.05 | 2.14 | 46.19 |
| Conv100 | 128 | 32 | 0.88 | 0.05 | 0.74 | 0.05 | 0.74 | 0.05 | 14.54 | 155.4 |
| Conv101 | 32 | 128 | 0.80 | 0.05 | 0.74 | 0.05 | 0.74 | 0.05 | 1.6 | 46.19 |
| Conv102 | 32 | 288 | 0.85 | 0.05 | 0.83 | 0.05 | 0.83 | 0.05 | 1.48 | 46.19 |
| Conv103 | 128 | 32 | 0.85 | 0.05 | 0.74 | 0.05 | 0.74 | 0.05 | 12.08 | 155.4 |
| Conv104 | 32 | 128 | 0.82 | 0.05 | 0.74 | 0.05 | 0.74 | 0.05 | 4.97 | 46.19 |
| Conv105 | 32 | 288 | 0.67 | 0.05 | 0.83 | 0.05 | 0.83 | 0.05 | 2.16 | 46.19 |
| Conv106 | 128 | 32 | 0.59 | 0.05 | 0.74 | 0.05 | 0.74 | 0.05 | 22.44 | 155.4 |
| Conv107 | 32 | 128 | 0.64 | 0.05 | 0.74 | 0.05 | 0.74 | 0.05 | 7.26 | 46.19 |
| Conv108 | 32 | 288 | 0.65 | 0.05 | 0.83 | 0.05 | 0.83 | 0.05 | 1.05 | 46.19 |
| Conv109 | 128 | 32 | 0.67 | 0.05 | 0.74 | 0.05 | 0.74 | 0.05 | 32.58 | 155.4 |
| Conv110 | 64 | 128 | 0.49 | 0.05 | 0.82 | 0.05 | 0.82 | 0.05 | 6.7 | 83.68 |
| Conv111 | 64 | 576 | 0.81 | 0.05 | 0.97 | 0.05 | 0.97 | 0.05 | 1.62 | 83.68 |
| Conv112 | 256 | 64 | 0.47 | 0.05 | 0.90 | 0.05 | 0.90 | 0.05 | 76.58 | 294.32 |
| Conv113 | 64 | 256 | 0.85 | 0.05 | 0.90 | 0.05 | 0.90 | 0.05 | 2.22 | 83.68 |
| Conv114 | 64 | 576 | 0.70 | 0.05 | 0.97 | 0.05 | 0.97 | 0.05 | 1.93 | 83.68 |
| Conv115 | 256 | 64 | 0.69 | 0.05 | 0.90 | 0.05 | 0.90 | 0.05 | 24.15 | 294.32 |
| Conv116 | 64 | 256 | 0.80 | 0.05 | 0.90 | 0.05 | 0.90 | 0.05 | 2.17 | 83.68 |
| Conv117 | 64 | 576 | 0.81 | 0.05 | 0.97 | 0.05 | 0.97 | 0.05 | 1.47 | 83.68 |
| Conv118 | 256 | 64 | 0.70 | 0.05 | 0.90 | 0.05 | 0.90 | 0.05 | 23.34 | 294.32 |
| Conv119 | 64 | 256 | 0.81 | 0.05 | 0.90 | 0.05 | 0.90 | 0.05 | 1.76 | 83.68 |
| Conv120 | 64 | 576 | 0.85 | 0.05 | 0.97 | 0.05 | 0.97 | 0.05 | 1.17 | 83.68 |
| Conv121 | 256 | 64 | 0.69 | 0.05 | 0.90 | 0.05 | 0.90 | 0.05 | 16.53 | 294.32 |
| Conv122 | 64 | 256 | 0.70 | 0.05 | 0.90 | 0.05 | 0.90 | 0.05 | 1.66 | 83.68 |
| Conv123 | 64 | 576 | 0.83 | 0.05 | 0.97 | 0.05 | 0.97 | 0.05 | 0.99 | 83.68 |
| Conv124 | 256 | 64 | 0.80 | 0.05 | 0.90 | 0.05 | 0.90 | 0.05 | 15.94 | 294.32 |
| Conv125 | 64 | 256 | 0.80 | 0.05 | 0.90 | 0.05 | 0.90 | 0.05 | 2.09 | 83.68 |
| Conv126 | 64 | 576 | 0.71 | 0.05 | 0.97 | 0.05 | 0.97 | 0.05 | 1.94 | 83.68 |
| Conv127 | 256 | 64 | 0.68 | 0.05 | 0.90 | 0.05 | 0.90 | 0.05 | 20.79 | 294.32 |
| Conv128 | 64 | 256 | 0.70 | 0.05 | 0.90 | 0.05 | 0.90 | 0.05 | 1.87 | 83.68 |
| Conv129 | 64 | 576 | 0.84 | 0.05 | 0.97 | 0.05 | 0.97 | 0.05 | 1.45 | 83.68 |
| Conv130 | 256 | 64 | 0.68 | 0.05 | 0.90 | 0.05 | 0.90 | 0.05 | 17.26 | 294.32 |
| Conv131 | 64 | 256 | 0.68 | 0.05 | 0.90 | 0.05 | 0.90 | 0.05 | 2.05 | 83.68 |
| Conv132 | 64 | 576 | 0.82 | 0.05 | 0.97 | 0.05 | 0.97 | 0.05 | 1.15 | 83.68 |
| Conv133 | 256 | 64 | 0.81 | 0.05 | 0.90 | 0.05 | 0.90 | 0.05 | 15.07 | 294.32 |
| Conv134 | 64 | 256 | 0.61 | 0.05 | 0.90 | 0.05 | 0.90 | 0.05 | 2.3 | 83.68 |
| Conv135 | 64 | 576 | 0.67 | 0.05 | 0.97 | 0.05 | 0.97 | 0.05 | 1.2 | 83.68 |
| Conv136 | 256 | 64 | 0.65 | 0.05 | 0.90 | 0.05 | 0.90 | 0.05 | 17.54 | 294.32 |
| Conv137 | 64 | 256 | 0.67 | 0.05 | 0.90 | 0.05 | 0.90 | 0.05 | 1.75 | 83.68 |
| Conv138 | 64 | 576 | 0.84 | 0.05 | 0.97 | 0.05 | 0.97 | 0.05 | 1.06 | 83.68 |

| Conv139 | 256 | 64 | 0.69 | 0.05 | 0.90 | 0.05 | 0.90 | 0.05 | 16.9 | 294.32 |
| Conv140 | 64 | 256 | 0.66 | 0.05 | 0.90 | 0.05 | 0.90 | 0.05 | 1.9 | 83.68 |
| Conv141 | 64 | 576 | 0.85 | 0.05 | 0.97 | 0.05 | 0.97 | 0.05 | 1.59 | 83.68 |
| Conv142 | 256 | 64 | 0.81 | 0.05 | 0.90 | 0.05 | 0.90 | 0.05 | 18.95 | 294.32 |
| Conv143 | 64 | 256 | 0.65 | 0.05 | 0.90 | 0.05 | 0.90 | 0.05 | 1.51 | 83.68 |
| Conv144 | 64 | 576 | 0.84 | 0.05 | 0.97 | 0.05 | 0.97 | 0.05 | 1.07 | 83.68 |
| Conv145 | 256 | 64 | 0.68 | 0.05 | 0.90 | 0.05 | 0.90 | 0.05 | 19.91 | 294.32 |
| Conv146 | 64 | 256 | 0.69 | 0.05 | 0.90 | 0.05 | 0.90 | 0.05 | 1.26 | 83.68 |
| Conv147 | 64 | 576 | 0.84 | 0.05 | 0.97 | 0.05 | 0.97 | 0.05 | 1.73 | 83.68 |
| Conv148 | 256 | 64 | 0.62 | 0.05 | 0.90 | 0.05 | 0.90 | 0.05 | 23.67 | 294.32 |
| Conv149 | 64 | 256 | 0.70 | 0.05 | 0.90 | 0.05 | 0.90 | 0.05 | 1.44 | 83.68 |
| Conv150 | 64 | 576 | 0.81 | 0.05 | 0.97 | 0.05 | 0.97 | 0.05 | 1.26 | 83.68 |
| Conv151 | 256 | 64 | 0.61 | 0.05 | 0.90 | 0.05 | 0.90 | 0.05 | 29.19 | 294.32 |
| Conv152 | 64 | 256 | 0.65 | 0.05 | 0.90 | 0.05 | 0.90 | 0.05 | 1.48 | 83.68 |
| Conv153 | 64 | 576 | 0.84 | 0.05 | 0.97 | 0.05 | 0.97 | 0.05 | 0.87 | 83.68 |
| Conv154 | 256 | 64 | 0.57 | 0.05 | 0.90 | 0.05 | 0.90 | 0.05 | 29.35 | 294.32 |
| Conv155 | 64 | 256 | 0.47 | 0.05 | 0.90 | 0.05 | 0.90 | 0.05 | 2.2 | 83.68 |
| Conv156 | 64 | 576 | 0.84 | 0.05 | 0.97 | 0.05 | 0.97 | 0.05 | 1.07 | 83.68 |
| Conv157 | 256 | 64 | 0.47 | 0.05 | 0.90 | 0.05 | 0.90 | 0.05 | 42.43 | 294.32 |
| Conv158 | 64 | 256 | 0.61 | 0.05 | 0.90 | 0.05 | 0.90 | 0.05 | 2.28 | 83.68 |
| Conv159 | 64 | 576 | 0.85 | 0.05 | 0.97 | 0.05 | 0.97 | 0.05 | 0.84 | 83.68 |
| Conv160 | 256 | 64 | 0.49 | 0.05 | 0.90 | 0.05 | 0.90 | 0.05 | 39.92 | 294.32 |
| Conv161 | 64 | 256 | 0.47 | 0.05 | 0.90 | 0.05 | 0.90 | 0.05 | 1.59 | 83.68 |
| Conv162 | 64 | 576 | 0.85 | 0.05 | 0.97 | 0.05 | 0.97 | 0.05 | 0.97 | 83.68 |
| Conv163 | 256 | 64 | 0.36 | 0.05 | 0.90 | 0.05 | 0.90 | 0.05 | 78.75 | 294.32 |
| **Passing rate** | - | - | 98.77% | | 100.0% | | 100.0% | | 96.32% | |

Table 34: Cifar100 Xavier-uniform-ResNet164

| Layer | Number | dim | Gaussian | | Mean_Left | | Mean_Right | | Sigma | |
|-------|--------|-----|---------|---------|---------|---------|---------|---------|---------|---------|
| | | | p-value | c-value | p-value | c-value | p-value | c-value | t-value | c-value |
| Conv1 | 16 | 27 | 0.00 | 0.05 | 0.58 | 0.05 | 0.42 | 0.05 | 5009.01 | 26.3 |
| Conv2 | 16 | 16 | 0.02 | 0.05 | 0.56 | 0.05 | 0.56 | 0.05 | 453.3 | 26.3 |
| Conv3 | 16 | 144 | 0.00 | 0.05 | 0.68 | 0.05 | 0.68 | 0.05 | 370.43 | 26.3 |
| Conv4 | 64 | 16 | 0.20 | 0.05 | 0.63 | 0.05 | 0.63 | 0.05 | 303.06 | 83.68 |
| Conv5 | 16 | 64 | 0.24 | 0.05 | 0.63 | 0.05 | 0.63 | 0.05 | 33.98 | 26.3 |
| Conv6 | 16 | 144 | 0.06 | 0.05 | 0.68 | 0.05 | 0.68 | 0.05 | 61.45 | 26.3 |
| Conv7 | 64 | 16 | 0.41 | 0.05 | 0.63 | 0.05 | 0.63 | 0.05 | 72.84 | 83.68 |
| Conv8 | 16 | 64 | 0.42 | 0.05 | 0.63 | 0.05 | 0.63 | 0.05 | 8.9 | 26.3 |
| Conv9 | 16 | 144 | 0.42 | 0.05 | 0.68 | 0.05 | 0.68 | 0.05 | 11.42 | 26.3 |
| Conv10 | 64 | 16 | 0.47 | 0.05 | 0.63 | 0.05 | 0.63 | 0.05 | 46.89 | 83.68 |
| Conv11 | 16 | 64 | 0.52 | 0.05 | 0.63 | 0.05 | 0.63 | 0.05 | 5.64 | 26.3 |
| Conv12 | 16 | 144 | 0.53 | 0.05 | 0.68 | 0.05 | 0.68 | 0.05 | 6.54 | 26.3 |
| Conv13 | 64 | 16 | 0.70 | 0.05 | 0.63 | 0.05 | 0.63 | 0.05 | 11.94 | 83.68 |
| Conv14 | 16 | 64 | 0.59 | 0.05 | 0.63 | 0.05 | 0.63 | 0.05 | 4.51 | 26.3 |
| Conv15 | 16 | 144 | 0.60 | 0.05 | 0.68 | 0.05 | 0.68 | 0.05 | 2.63 | 26.3 |
| Conv16 | 64 | 16 | 0.67 | 0.05 | 0.63 | 0.05 | 0.63 | 0.05 | 13.71 | 83.68 |
| Conv17 | 16 | 64 | 0.70 | 0.05 | 0.63 | 0.05 | 0.63 | 0.05 | 2.34 | 26.3 |
| Conv18 | 16 | 144 | 0.68 | 0.05 | 0.68 | 0.05 | 0.68 | 0.05 | 1.4 | 26.3 |
| Conv19 | 64 | 16 | 0.85 | 0.05 | 0.63 | 0.05 | 0.63 | 0.05 | 7.47 | 83.68 |
| Conv20 | 16 | 64 | 0.52 | 0.05 | 0.63 | 0.05 | 0.63 | 0.05 | 6.4 | 26.3 |
| Conv21 | 16 | 144 | 0.54 | 0.05 | 0.68 | 0.05 | 0.68 | 0.05 | 3.57 | 26.3 |
| Conv22 | 64 | 16 | 0.63 | 0.05 | 0.63 | 0.05 | 0.63 | 0.05 | 22.62 | 83.68 |
| Conv23 | 16 | 64 | 0.49 | 0.05 | 0.63 | 0.05 | 0.63 | 0.05 | 9.23 | 26.3 |
| Conv24 | 16 | 144 | 0.47 | 0.05 | 0.68 | 0.05 | 0.68 | 0.05 | 3.77 | 26.3 |

| | | | | | | | | | | |
|---|---|---|---|---|---|---|---|---|---|---|
| Conv25 | 64 | 16 | 0.65 | 0.05 | 0.63 | 0.05 | 0.63 | 0.05 | 18.83 | 83.68 |
| Conv26 | 16 | 64 | 0.52 | 0.05 | 0.63 | 0.05 | 0.63 | 0.05 | 3.82 | 26.3 |
| Conv27 | 16 | 144 | 0.84 | 0.05 | 0.68 | 0.05 | 0.68 | 0.05 | 0.88 | 26.3 |
| Conv28 | 64 | 16 | 0.82 | 0.05 | 0.63 | 0.05 | 0.63 | 0.05 | 9.15 | 83.68 |
| Conv29 | 16 | 64 | 0.61 | 0.05 | 0.63 | 0.05 | 0.63 | 0.05 | 5.81 | 26.3 |
| Conv30 | 16 | 144 | 0.56 | 0.05 | 0.68 | 0.05 | 0.68 | 0.05 | 3.08 | 26.3 |
| Conv31 | 64 | 16 | 0.85 | 0.05 | 0.63 | 0.05 | 0.63 | 0.05 | 10.1 | 83.68 |
| Conv32 | 16 | 64 | 0.52 | 0.05 | 0.63 | 0.05 | 0.63 | 0.05 | 8.13 | 26.3 |
| Conv33 | 16 | 144 | 0.49 | 0.05 | 0.68 | 0.05 | 0.68 | 0.05 | 4.66 | 26.3 |
| Conv34 | 64 | 16 | 0.63 | 0.05 | 0.63 | 0.05 | 0.63 | 0.05 | 14.52 | 83.68 |
| Conv35 | 16 | 64 | 0.70 | 0.05 | 0.63 | 0.05 | 0.63 | 0.05 | 2.11 | 26.3 |
| Conv36 | 16 | 144 | 0.86 | 0.05 | 0.68 | 0.05 | 0.68 | 0.05 | 1.17 | 26.3 |
| Conv37 | 64 | 16 | 0.66 | 0.05 | 0.63 | 0.05 | 0.63 | 0.05 | 9.5 | 83.68 |
| Conv38 | 16 | 64 | 0.59 | 0.05 | 0.63 | 0.05 | 0.63 | 0.05 | 7.3 | 26.3 |
| Conv39 | 16 | 144 | 0.67 | 0.05 | 0.68 | 0.05 | 0.68 | 0.05 | 2.4 | 26.3 |
| Conv40 | 64 | 16 | 0.80 | 0.05 | 0.63 | 0.05 | 0.63 | 0.05 | 9.38 | 83.68 |
| Conv41 | 16 | 64 | 0.84 | 0.05 | 0.63 | 0.05 | 0.63 | 0.05 | 2.45 | 26.3 |
| Conv42 | 16 | 144 | 0.85 | 0.05 | 0.68 | 0.05 | 0.68 | 0.05 | 2.08 | 26.3 |
| Conv43 | 64 | 16 | 0.84 | 0.05 | 0.63 | 0.05 | 0.63 | 0.05 | 9.47 | 83.68 |
| Conv44 | 16 | 64 | 0.38 | 0.05 | 0.63 | 0.05 | 0.63 | 0.05 | 15.34 | 26.3 |
| Conv45 | 16 | 144 | 0.38 | 0.05 | 0.68 | 0.05 | 0.68 | 0.05 | 6.86 | 26.3 |
| Conv46 | 64 | 16 | 0.66 | 0.05 | 0.63 | 0.05 | 0.63 | 0.05 | 20.76 | 83.68 |
| Conv47 | 16 | 64 | 0.53 | 0.05 | 0.63 | 0.05 | 0.63 | 0.05 | 10.64 | 26.3 |
| Conv48 | 16 | 144 | 0.67 | 0.05 | 0.68 | 0.05 | 0.68 | 0.05 | 3.99 | 26.3 |
| Conv49 | 64 | 16 | 0.67 | 0.05 | 0.63 | 0.05 | 0.63 | 0.05 | 19.94 | 83.68 |
| Conv50 | 16 | 64 | 0.50 | 0.05 | 0.63 | 0.05 | 0.63 | 0.05 | 12.16 | 26.3 |
| Conv51 | 16 | 144 | 0.57 | 0.05 | 0.68 | 0.05 | 0.68 | 0.05 | 2.76 | 26.3 |
| Conv52 | 64 | 16 | 0.66 | 0.05 | 0.63 | 0.05 | 0.63 | 0.05 | 21.25 | 83.68 |
| Conv53 | 16 | 64 | 0.59 | 0.05 | 0.63 | 0.05 | 0.63 | 0.05 | 6.63 | 26.3 |
| Conv54 | 16 | 144 | 0.81 | 0.05 | 0.68 | 0.05 | 0.68 | 0.05 | 2.3 | 26.3 |
| Conv55 | 64 | 16 | 0.82 | 0.05 | 0.63 | 0.05 | 0.63 | 0.05 | 14.84 | 83.68 |
| Conv56 | 32 | 64 | 0.52 | 0.05 | 0.67 | 0.05 | 0.67 | 0.05 | 15.63 | 46.19 |
| Conv57 | 32 | 288 | 0.63 | 0.05 | 0.83 | 0.05 | 0.83 | 0.05 | 3.06 | 46.19 |
| Conv58 | 128 | 32 | 0.52 | 0.05 | 0.74 | 0.05 | 0.74 | 0.05 | 52.26 | 155.4 |
| Conv59 | 32 | 128 | 0.82 | 0.05 | 0.74 | 0.05 | 0.74 | 0.05 | 4.0 | 46.19 |
| Conv60 | 32 | 288 | 0.61 | 0.05 | 0.83 | 0.05 | 0.83 | 0.05 | 4.98 | 46.19 |
| Conv61 | 128 | 32 | 0.66 | 0.05 | 0.74 | 0.05 | 0.74 | 0.05 | 28.38 | 155.4 |
| Conv62 | 32 | 128 | 0.68 | 0.05 | 0.74 | 0.05 | 0.74 | 0.05 | 3.28 | 46.19 |
| Conv63 | 32 | 288 | 0.66 | 0.05 | 0.83 | 0.05 | 0.83 | 0.05 | 2.88 | 46.19 |
| Conv64 | 128 | 32 | 0.82 | 0.05 | 0.74 | 0.05 | 0.74 | 0.05 | 13.36 | 155.4 |
| Conv65 | 32 | 128 | 0.80 | 0.05 | 0.74 | 0.05 | 0.74 | 0.05 | 3.11 | 46.19 |
| Conv66 | 32 | 288 | 0.70 | 0.05 | 0.83 | 0.05 | 0.83 | 0.05 | 1.08 | 46.19 |
| Conv67 | 128 | 32 | 0.80 | 0.05 | 0.74 | 0.05 | 0.74 | 0.05 | 11.6 | 155.4 |
| Conv68 | 32 | 128 | 0.86 | 0.05 | 0.74 | 0.05 | 0.74 | 0.05 | 1.2 | 46.19 |
| Conv69 | 32 | 288 | 0.81 | 0.05 | 0.83 | 0.05 | 0.83 | 0.05 | 1.38 | 46.19 |
| Conv70 | 128 | 32 | 0.85 | 0.05 | 0.74 | 0.05 | 0.74 | 0.05 | 13.39 | 155.4 |
| Conv71 | 32 | 128 | 0.86 | 0.05 | 0.74 | 0.05 | 0.74 | 0.05 | 2.05 | 46.19 |
| Conv72 | 32 | 288 | 0.82 | 0.05 | 0.83 | 0.05 | 0.83 | 0.05 | 2.02 | 46.19 |
| Conv73 | 128 | 32 | 0.70 | 0.05 | 0.74 | 0.05 | 0.74 | 0.05 | 14.13 | 155.4 |
| Conv74 | 32 | 128 | 0.87 | 0.05 | 0.74 | 0.05 | 0.74 | 0.05 | 1.56 | 46.19 |
| Conv75 | 32 | 288 | 0.85 | 0.05 | 0.83 | 0.05 | 0.83 | 0.05 | 1.73 | 46.19 |
| Conv76 | 128 | 32 | 0.83 | 0.05 | 0.74 | 0.05 | 0.74 | 0.05 | 12.27 | 155.4 |
| Conv77 | 32 | 128 | 0.80 | 0.05 | 0.74 | 0.05 | 0.74 | 0.05 | 2.22 | 46.19 |
| Conv78 | 32 | 288 | 0.63 | 0.05 | 0.83 | 0.05 | 0.83 | 0.05 | 2.79 | 46.19 |
| Conv79 | 128 | 32 | 0.67 | 0.05 | 0.74 | 0.05 | 0.74 | 0.05 | 11.46 | 155.4 |
| Conv80 | 32 | 128 | 0.50 | 0.05 | 0.74 | 0.05 | 0.74 | 0.05 | 5.83 | 46.19 |
| Conv81 | 32 | 288 | 0.65 | 0.05 | 0.83 | 0.05 | 0.83 | 0.05 | 2.65 | 46.19 |
| Conv82 | 128 | 32 | 0.83 | 0.05 | 0.74 | 0.05 | 0.74 | 0.05 | 18.09 | 155.4 |
| Conv83 | 32 | 128 | 0.80 | 0.05 | 0.74 | 0.05 | 0.74 | 0.05 | 3.61 | 46.19 |

| | | | | | | | | | | |
|---|---|---|---|---|---|---|---|---|---|---|
| Conv84 | 32 | 288 | 0.87 | 0.05 | 0.83 | 0.05 | 0.83 | 0.05 | 1.97 | 46.19 |
| Conv85 | 128 | 32 | 0.64 | 0.05 | 0.74 | 0.05 | 0.74 | 0.05 | 21.99 | 155.4 |
| Conv86 | 32 | 128 | 0.87 | 0.05 | 0.74 | 0.05 | 0.74 | 0.05 | 2.38 | 46.19 |
| Conv87 | 32 | 288 | 0.81 | 0.05 | 0.83 | 0.05 | 0.83 | 0.05 | 1.85 | 46.19 |
| Conv88 | 128 | 32 | 0.87 | 0.05 | 0.74 | 0.05 | 0.74 | 0.05 | 14.13 | 155.4 |
| Conv89 | 32 | 128 | 0.85 | 0.05 | 0.74 | 0.05 | 0.74 | 0.05 | 2.08 | 46.19 |
| Conv90 | 32 | 288 | 0.87 | 0.05 | 0.83 | 0.05 | 0.83 | 0.05 | 1.11 | 46.19 |
| Conv91 | 128 | 32 | 0.85 | 0.05 | 0.74 | 0.05 | 0.74 | 0.05 | 15.24 | 155.4 |
| Conv92 | 32 | 128 | 0.83 | 0.05 | 0.74 | 0.05 | 0.74 | 0.05 | 2.64 | 46.19 |
| Conv93 | 32 | 288 | 0.83 | 0.05 | 0.83 | 0.05 | 0.83 | 0.05 | 1.89 | 46.19 |
| Conv94 | 128 | 32 | 0.83 | 0.05 | 0.74 | 0.05 | 0.74 | 0.05 | 15.49 | 155.4 |
| Conv95 | 32 | 128 | 0.82 | 0.05 | 0.74 | 0.05 | 0.74 | 0.05 | 3.04 | 46.19 |
| Conv96 | 32 | 288 | 0.84 | 0.05 | 0.83 | 0.05 | 0.83 | 0.05 | 1.86 | 46.19 |
| Conv97 | 128 | 32 | 0.84 | 0.05 | 0.74 | 0.05 | 0.74 | 0.05 | 16.24 | 155.4 |
| Conv98 | 32 | 128 | 0.55 | 0.05 | 0.74 | 0.05 | 0.74 | 0.05 | 6.03 | 46.19 |
| Conv99 | 32 | 288 | 0.62 | 0.05 | 0.83 | 0.05 | 0.83 | 0.05 | 4.51 | 46.19 |
| Conv100 | 128 | 32 | 0.59 | 0.05 | 0.74 | 0.05 | 0.74 | 0.05 | 24.42 | 155.4 |
| Conv101 | 32 | 128 | 0.57 | 0.05 | 0.74 | 0.05 | 0.74 | 0.05 | 6.83 | 46.19 |
| Conv102 | 32 | 288 | 0.70 | 0.05 | 0.83 | 0.05 | 0.83 | 0.05 | 2.47 | 46.19 |
| Conv103 | 128 | 32 | 0.71 | 0.05 | 0.74 | 0.05 | 0.74 | 0.05 | 25.11 | 155.4 |
| Conv104 | 32 | 128 | 0.83 | 0.05 | 0.74 | 0.05 | 0.74 | 0.05 | 2.85 | 46.19 |
| Conv105 | 32 | 288 | 0.85 | 0.05 | 0.83 | 0.05 | 0.83 | 0.05 | 0.99 | 46.19 |
| Conv106 | 128 | 32 | 0.81 | 0.05 | 0.74 | 0.05 | 0.74 | 0.05 | 16.7 | 155.4 |
| Conv107 | 32 | 128 | 0.61 | 0.05 | 0.74 | 0.05 | 0.74 | 0.05 | 3.72 | 46.19 |
| Conv108 | 32 | 288 | 0.85 | 0.05 | 0.83 | 0.05 | 0.83 | 0.05 | 1.75 | 46.19 |
| Conv109 | 128 | 32 | 0.64 | 0.05 | 0.74 | 0.05 | 0.74 | 0.05 | 22.31 | 155.4 |
| Conv110 | 64 | 128 | 0.55 | 0.05 | 0.82 | 0.05 | 0.82 | 0.05 | 9.69 | 83.68 |
| Conv111 | 64 | 576 | 0.81 | 0.05 | 0.97 | 0.05 | 0.97 | 0.05 | 1.92 | 83.68 |
| Conv112 | 256 | 64 | 0.52 | 0.05 | 0.90 | 0.05 | 0.90 | 0.05 | 70.08 | 294.32 |
| Conv113 | 64 | 256 | 0.62 | 0.05 | 0.90 | 0.05 | 0.90 | 0.05 | 1.6 | 83.68 |
| Conv114 | 64 | 576 | 0.84 | 0.05 | 0.97 | 0.05 | 0.97 | 0.05 | 2.01 | 83.68 |
| Conv115 | 256 | 64 | 0.59 | 0.05 | 0.90 | 0.05 | 0.90 | 0.05 | 17.41 | 294.32 |
| Conv116 | 64 | 256 | 0.81 | 0.05 | 0.90 | 0.05 | 0.90 | 0.05 | 1.63 | 83.68 |
| Conv117 | 64 | 576 | 0.84 | 0.05 | 0.97 | 0.05 | 0.97 | 0.05 | 1.61 | 83.68 |
| Conv118 | 256 | 64 | 0.66 | 0.05 | 0.90 | 0.05 | 0.90 | 0.05 | 12.81 | 294.32 |
| Conv119 | 64 | 256 | 0.67 | 0.05 | 0.90 | 0.05 | 0.90 | 0.05 | 2.57 | 83.68 |
| Conv120 | 64 | 576 | 0.68 | 0.05 | 0.97 | 0.05 | 0.97 | 0.05 | 2.84 | 83.68 |
| Conv121 | 256 | 64 | 0.68 | 0.05 | 0.90 | 0.05 | 0.90 | 0.05 | 17.96 | 294.32 |
| Conv122 | 64 | 256 | 0.70 | 0.05 | 0.90 | 0.05 | 0.90 | 0.05 | 2.2 | 83.68 |
| Conv123 | 64 | 576 | 0.81 | 0.05 | 0.97 | 0.05 | 0.97 | 0.05 | 1.95 | 83.68 |
| Conv124 | 256 | 64 | 0.69 | 0.05 | 0.90 | 0.05 | 0.90 | 0.05 | 19.88 | 294.32 |
| Conv125 | 64 | 256 | 0.63 | 0.05 | 0.90 | 0.05 | 0.90 | 0.05 | 1.32 | 83.68 |
| Conv126 | 64 | 576 | 0.86 | 0.05 | 0.97 | 0.05 | 0.97 | 0.05 | 0.81 | 83.68 |
| Conv127 | 256 | 64 | 0.63 | 0.05 | 0.90 | 0.05 | 0.90 | 0.05 | 15.33 | 294.32 |
| Conv128 | 64 | 256 | 0.59 | 0.05 | 0.90 | 0.05 | 0.90 | 0.05 | 2.0 | 83.68 |
| Conv129 | 64 | 576 | 0.81 | 0.05 | 0.97 | 0.05 | 0.97 | 0.05 | 1.35 | 83.68 |
| Conv130 | 256 | 64 | 0.67 | 0.05 | 0.90 | 0.05 | 0.90 | 0.05 | 14.09 | 294.32 |
| Conv131 | 64 | 256 | 0.54 | 0.05 | 0.90 | 0.05 | 0.90 | 0.05 | 1.88 | 83.68 |
| Conv132 | 64 | 576 | 0.85 | 0.05 | 0.97 | 0.05 | 0.97 | 0.05 | 1.32 | 83.68 |
| Conv133 | 256 | 64 | 0.67 | 0.05 | 0.90 | 0.05 | 0.90 | 0.05 | 19.46 | 294.32 |
| Conv134 | 64 | 256 | 0.63 | 0.05 | 0.90 | 0.05 | 0.90 | 0.05 | 1.51 | 83.68 |
| Conv135 | 64 | 576 | 0.84 | 0.05 | 0.97 | 0.05 | 0.97 | 0.05 | 0.92 | 83.68 |
| Conv136 | 256 | 64 | 0.65 | 0.05 | 0.90 | 0.05 | 0.90 | 0.05 | 18.41 | 294.32 |
| Conv137 | 64 | 256 | 0.66 | 0.05 | 0.90 | 0.05 | 0.90 | 0.05 | 1.55 | 83.68 |
| Conv138 | 64 | 576 | 0.85 | 0.05 | 0.97 | 0.05 | 0.97 | 0.05 | 1.18 | 83.68 |
| Conv139 | 256 | 64 | 0.65 | 0.05 | 0.90 | 0.05 | 0.90 | 0.05 | 21.62 | 294.32 |
| Conv140 | 64 | 256 | 0.80 | 0.05 | 0.90 | 0.05 | 0.90 | 0.05 | 1.64 | 83.68 |
| Conv141 | 64 | 576 | 0.84 | 0.05 | 0.97 | 0.05 | 0.97 | 0.05 | 0.78 | 83.68 |
| Conv142 | 256 | 64 | 0.63 | 0.05 | 0.90 | 0.05 | 0.90 | 0.05 | 23.34 | 294.32 |

| | | | Gaussian | | Mean_Left | | Mean_Right | | Sigma | |
|---|---|---|---|---|---|---|---|---|---|---|
| Conv143 | 64 | 256 | 0.62 | 0.05 | 0.90 | 0.05 | 0.90 | 0.05 | 1.68 | 83.68 |
| Conv144 | 64 | 576 | 0.83 | 0.05 | 0.97 | 0.05 | 0.97 | 0.05 | 0.95 | 83.68 |
| Conv145 | 256 | 64 | 0.67 | 0.05 | 0.90 | 0.05 | 0.90 | 0.05 | 25.4 | 294.32 |
| Conv146 | 64 | 256 | 0.69 | 0.05 | 0.90 | 0.05 | 0.90 | 0.05 | 1.81 | 83.68 |
| Conv147 | 64 | 576 | 0.85 | 0.05 | 0.97 | 0.05 | 0.97 | 0.05 | 0.92 | 83.68 |
| Conv148 | 256 | 64 | 0.52 | 0.05 | 0.90 | 0.05 | 0.90 | 0.05 | 30.63 | 294.32 |
| Conv149 | 64 | 256 | 0.62 | 0.05 | 0.90 | 0.05 | 0.90 | 0.05 | 2.0 | 83.68 |
| Conv150 | 64 | 576 | 0.86 | 0.05 | 0.97 | 0.05 | 0.97 | 0.05 | 0.84 | 83.68 |
| Conv151 | 256 | 64 | 0.56 | 0.05 | 0.90 | 0.05 | 0.90 | 0.05 | 33.48 | 294.32 |
| Conv152 | 64 | 256 | 0.63 | 0.05 | 0.90 | 0.05 | 0.90 | 0.05 | 1.26 | 83.68 |
| Conv153 | 64 | 576 | 0.86 | 0.05 | 0.97 | 0.05 | 0.97 | 0.05 | 0.69 | 83.68 |
| Conv154 | 256 | 64 | 0.51 | 0.05 | 0.90 | 0.05 | 0.90 | 0.05 | 37.11 | 294.32 |
| Conv155 | 64 | 256 | 0.66 | 0.05 | 0.90 | 0.05 | 0.90 | 0.05 | 1.72 | 83.68 |
| Conv156 | 64 | 576 | 0.86 | 0.05 | 0.97 | 0.05 | 0.97 | 0.05 | 0.65 | 83.68 |
| Conv157 | 256 | 64 | 0.62 | 0.05 | 0.90 | 0.05 | 0.90 | 0.05 | 28.49 | 294.32 |
| Conv158 | 64 | 256 | 0.58 | 0.05 | 0.90 | 0.05 | 0.90 | 0.05 | 1.7 | 83.68 |
| Conv159 | 64 | 576 | 0.86 | 0.05 | 0.97 | 0.05 | 0.97 | 0.05 | 0.82 | 83.68 |
| Conv160 | 256 | 64 | 0.57 | 0.05 | 0.90 | 0.05 | 0.90 | 0.05 | 42.43 | 294.32 |
| Conv161 | 64 | 256 | 0.53 | 0.05 | 0.90 | 0.05 | 0.90 | 0.05 | 2.0 | 83.68 |
| Conv162 | 64 | 576 | 0.87 | 0.05 | 0.97 | 0.05 | 0.97 | 0.05 | 0.51 | 83.68 |
| Conv163 | 256 | 64 | 0.54 | 0.05 | 0.90 | 0.05 | 0.90 | 0.05 | 45.77 | 294.32 |
| **Passing rate** | - | - | 98.16% | | 100.0% | | 100.0% | | 96.32% | |

Table 35: Cifar100 orthogonal-ResNet164

| Layer | Number | dim | Gaussian | | Mean_Left | | Mean_Right | | Sigma | |
|---|---|---|---|---|---|---|---|---|---|---|
| | | | p-value | c-value | p-value | c-value | p-value | c-value | t-value | c-value |
| Conv1 | 16 | 27 | 0.00 | 0.05 | 0.58 | 0.05 | 0.42 | 0.05 | 1011.26 | 26.3 |
| Conv2 | 16 | 16 | 0.02 | 0.05 | 0.56 | 0.05 | 0.56 | 0.05 | 510.87 | 26.3 |
| Conv3 | 16 | 144 | 0.05 | 0.05 | 0.68 | 0.05 | 0.68 | 0.05 | 91.21 | 26.3 |
| Conv4 | 64 | 16 | 0.01 | 0.05 | 0.63 | 0.05 | 0.63 | 0.05 | 837.24 | 83.68 |
| Conv5 | 16 | 64 | 0.02 | 0.05 | 0.63 | 0.05 | 0.63 | 0.05 | 132.43 | 26.3 |
| Conv6 | 16 | 144 | 0.17 | 0.05 | 0.68 | 0.05 | 0.68 | 0.05 | 14.32 | 26.3 |
| Conv7 | 64 | 16 | 0.19 | 0.05 | 0.63 | 0.05 | 0.63 | 0.05 | 213.05 | 83.68 |
| Conv8 | 16 | 64 | 0.29 | 0.05 | 0.63 | 0.05 | 0.63 | 0.05 | 25.29 | 26.3 |
| Conv9 | 16 | 144 | 0.37 | 0.05 | 0.68 | 0.05 | 0.68 | 0.05 | 11.98 | 26.3 |
| Conv10 | 64 | 16 | 0.46 | 0.05 | 0.63 | 0.05 | 0.63 | 0.05 | 59.8 | 83.68 |
| Conv11 | 16 | 64 | 0.40 | 0.05 | 0.63 | 0.05 | 0.63 | 0.05 | 8.49 | 26.3 |
| Conv12 | 16 | 144 | 0.50 | 0.05 | 0.68 | 0.05 | 0.68 | 0.05 | 8.69 | 26.3 |
| Conv13 | 64 | 16 | 0.31 | 0.05 | 0.63 | 0.05 | 0.63 | 0.05 | 59.75 | 83.68 |
| Conv14 | 16 | 64 | 0.44 | 0.05 | 0.63 | 0.05 | 0.63 | 0.05 | 14.31 | 26.3 |
| Conv15 | 16 | 144 | 0.67 | 0.05 | 0.68 | 0.05 | 0.68 | 0.05 | 1.84 | 26.3 |
| Conv16 | 64 | 16 | 0.65 | 0.05 | 0.63 | 0.05 | 0.63 | 0.05 | 20.38 | 83.68 |
| Conv17 | 16 | 64 | 0.59 | 0.05 | 0.63 | 0.05 | 0.63 | 0.05 | 2.67 | 26.3 |
| Conv18 | 16 | 144 | 0.64 | 0.05 | 0.68 | 0.05 | 0.68 | 0.05 | 2.39 | 26.3 |
| Conv19 | 64 | 16 | 0.65 | 0.05 | 0.63 | 0.05 | 0.63 | 0.05 | 11.96 | 83.68 |
| Conv20 | 16 | 64 | 0.68 | 0.05 | 0.63 | 0.05 | 0.63 | 0.05 | 2.11 | 26.3 |
| Conv21 | 16 | 144 | 0.84 | 0.05 | 0.68 | 0.05 | 0.68 | 0.05 | 0.51 | 26.3 |
| Conv22 | 64 | 16 | 0.86 | 0.05 | 0.63 | 0.05 | 0.63 | 0.05 | 6.61 | 83.68 |
| Conv23 | 16 | 64 | 0.68 | 0.05 | 0.63 | 0.05 | 0.63 | 0.05 | 2.92 | 26.3 |
| Conv24 | 16 | 144 | 0.83 | 0.05 | 0.68 | 0.05 | 0.68 | 0.05 | 0.98 | 26.3 |
| Conv25 | 64 | 16 | 0.80 | 0.05 | 0.63 | 0.05 | 0.63 | 0.05 | 7.3 | 83.68 |
| Conv26 | 16 | 64 | 0.83 | 0.05 | 0.63 | 0.05 | 0.63 | 0.05 | 1.15 | 26.3 |
| Conv27 | 16 | 144 | 0.82 | 0.05 | 0.68 | 0.05 | 0.68 | 0.05 | 0.61 | 26.3 |
| Conv28 | 64 | 16 | 0.88 | 0.05 | 0.63 | 0.05 | 0.63 | 0.05 | 3.66 | 83.68 |

| | | | | | | | | | | | |
|---|---|---|---|---|---|---|---|---|---|---|---|
| Conv29 | 16 | 64 | 0.83 | 0.05 | 0.63 | 0.05 | 0.63 | 0.05 | 1.14 | 26.3 |
| Conv30 | 16 | 144 | 0.87 | 0.05 | 0.68 | 0.05 | 0.68 | 0.05 | 0.49 | 26.3 |
| Conv31 | 64 | 16 | 0.88 | 0.05 | 0.63 | 0.05 | 0.63 | 0.05 | 4.51 | 83.68 |
| Conv32 | 16 | 64 | 0.64 | 0.05 | 0.63 | 0.05 | 0.63 | 0.05 | 4.88 | 26.3 |
| Conv33 | 16 | 144 | 0.40 | 0.05 | 0.68 | 0.05 | 0.68 | 0.05 | 4.29 | 26.3 |
| Conv34 | 64 | 16 | 0.80 | 0.05 | 0.63 | 0.05 | 0.63 | 0.05 | 11.26 | 83.68 |
| Conv35 | 16 | 64 | 0.66 | 0.05 | 0.63 | 0.05 | 0.63 | 0.05 | 2.42 | 26.3 |
| Conv36 | 16 | 144 | 0.83 | 0.05 | 0.68 | 0.05 | 0.68 | 0.05 | 0.4 | 26.3 |
| Conv37 | 64 | 16 | 0.88 | 0.05 | 0.63 | 0.05 | 0.63 | 0.05 | 2.93 | 83.68 |
| Conv38 | 16 | 64 | 0.67 | 0.05 | 0.63 | 0.05 | 0.63 | 0.05 | 3.22 | 26.3 |
| Conv39 | 16 | 144 | 0.88 | 0.05 | 0.68 | 0.05 | 0.68 | 0.05 | 0.64 | 26.3 |
| Conv40 | 64 | 16 | 0.86 | 0.05 | 0.63 | 0.05 | 0.63 | 0.05 | 4.46 | 83.68 |
| Conv41 | 16 | 64 | 0.67 | 0.05 | 0.63 | 0.05 | 0.63 | 0.05 | 4.24 | 26.3 |
| Conv42 | 16 | 144 | 0.67 | 0.05 | 0.68 | 0.05 | 0.68 | 0.05 | 2.75 | 26.3 |
| Conv43 | 64 | 16 | 0.63 | 0.05 | 0.63 | 0.05 | 0.63 | 0.05 | 11.6 | 83.68 |
| Conv44 | 16 | 64 | 0.70 | 0.05 | 0.63 | 0.05 | 0.63 | 0.05 | 2.71 | 26.3 |
| Conv45 | 16 | 144 | 0.85 | 0.05 | 0.68 | 0.05 | 0.68 | 0.05 | 0.51 | 26.3 |
| Conv46 | 64 | 16 | 0.89 | 0.05 | 0.63 | 0.05 | 0.63 | 0.05 | 5.22 | 83.68 |
| Conv47 | 16 | 64 | 0.55 | 0.05 | 0.63 | 0.05 | 0.63 | 0.05 | 7.37 | 26.3 |
| Conv48 | 16 | 144 | 0.63 | 0.05 | 0.68 | 0.05 | 0.68 | 0.05 | 3.1 | 26.3 |
| Conv49 | 64 | 16 | 0.82 | 0.05 | 0.63 | 0.05 | 0.63 | 0.05 | 14.35 | 83.68 |
| Conv50 | 16 | 64 | 0.53 | 0.05 | 0.63 | 0.05 | 0.63 | 0.05 | 7.6 | 26.3 |
| Conv51 | 16 | 144 | 0.63 | 0.05 | 0.68 | 0.05 | 0.68 | 0.05 | 4.49 | 26.3 |
| Conv52 | 64 | 16 | 0.81 | 0.05 | 0.63 | 0.05 | 0.63 | 0.05 | 14.16 | 83.68 |
| Conv53 | 16 | 64 | 0.70 | 0.05 | 0.63 | 0.05 | 0.63 | 0.05 | 5.13 | 26.3 |
| Conv54 | 16 | 144 | 0.68 | 0.05 | 0.68 | 0.05 | 0.68 | 0.05 | 1.98 | 26.3 |
| Conv55 | 64 | 16 | 0.69 | 0.05 | 0.63 | 0.05 | 0.63 | 0.05 | 10.12 | 83.68 |
| Conv56 | 32 | 64 | 0.47 | 0.05 | 0.67 | 0.05 | 0.67 | 0.05 | 24.35 | 46.19 |
| Conv57 | 32 | 288 | 0.53 | 0.05 | 0.83 | 0.05 | 0.83 | 0.05 | 7.11 | 46.19 |
| Conv58 | 128 | 32 | 0.39 | 0.05 | 0.74 | 0.05 | 0.74 | 0.05 | 130.03 | 155.4 |
| Conv59 | 32 | 128 | 0.64 | 0.05 | 0.74 | 0.05 | 0.74 | 0.05 | 4.38 | 46.19 |
| Conv60 | 32 | 288 | 0.53 | 0.05 | 0.83 | 0.05 | 0.83 | 0.05 | 5.98 | 46.19 |
| Conv61 | 128 | 32 | 0.66 | 0.05 | 0.74 | 0.05 | 0.74 | 0.05 | 46.94 | 155.4 |
| Conv62 | 32 | 128 | 0.65 | 0.05 | 0.74 | 0.05 | 0.74 | 0.05 | 3.71 | 46.19 |
| Conv63 | 32 | 288 | 0.68 | 0.05 | 0.83 | 0.05 | 0.83 | 0.05 | 1.96 | 46.19 |
| Conv64 | 128 | 32 | 0.67 | 0.05 | 0.74 | 0.05 | 0.74 | 0.05 | 22.74 | 155.4 |
| Conv65 | 32 | 128 | 0.68 | 0.05 | 0.74 | 0.05 | 0.74 | 0.05 | 4.41 | 46.19 |
| Conv66 | 32 | 288 | 0.65 | 0.05 | 0.83 | 0.05 | 0.83 | 0.05 | 4.22 | 46.19 |
| Conv67 | 128 | 32 | 0.57 | 0.05 | 0.74 | 0.05 | 0.74 | 0.05 | 31.38 | 155.4 |
| Conv68 | 32 | 128 | 0.81 | 0.05 | 0.74 | 0.05 | 0.74 | 0.05 | 3.85 | 46.19 |
| Conv69 | 32 | 288 | 0.60 | 0.05 | 0.83 | 0.05 | 0.83 | 0.05 | 3.02 | 46.19 |
| Conv70 | 128 | 32 | 0.68 | 0.05 | 0.74 | 0.05 | 0.74 | 0.05 | 21.04 | 155.4 |
| Conv71 | 32 | 128 | 0.44 | 0.05 | 0.74 | 0.05 | 0.74 | 0.05 | 14.29 | 46.19 |
| Conv72 | 32 | 288 | 0.65 | 0.05 | 0.83 | 0.05 | 0.83 | 0.05 | 5.11 | 46.19 |
| Conv73 | 128 | 32 | 0.51 | 0.05 | 0.74 | 0.05 | 0.74 | 0.05 | 58.28 | 155.4 |
| Conv74 | 32 | 128 | 0.61 | 0.05 | 0.74 | 0.05 | 0.74 | 0.05 | 3.79 | 46.19 |
| Conv75 | 32 | 288 | 0.84 | 0.05 | 0.83 | 0.05 | 0.83 | 0.05 | 3.0 | 46.19 |
| Conv76 | 128 | 32 | 0.83 | 0.05 | 0.74 | 0.05 | 0.74 | 0.05 | 12.29 | 155.4 |
| Conv77 | 32 | 128 | 0.85 | 0.05 | 0.74 | 0.05 | 0.74 | 0.05 | 2.66 | 46.19 |
| Conv78 | 32 | 288 | 0.82 | 0.05 | 0.83 | 0.05 | 0.83 | 0.05 | 2.3 | 46.19 |
| Conv79 | 128 | 32 | 0.85 | 0.05 | 0.74 | 0.05 | 0.74 | 0.05 | 13.49 | 155.4 |
| Conv80 | 32 | 128 | 0.61 | 0.05 | 0.74 | 0.05 | 0.74 | 0.05 | 5.51 | 46.19 |
| Conv81 | 32 | 288 | 0.84 | 0.05 | 0.83 | 0.05 | 0.83 | 0.05 | 2.35 | 46.19 |
| Conv82 | 128 | 32 | 0.71 | 0.05 | 0.74 | 0.05 | 0.74 | 0.05 | 18.02 | 155.4 |
| Conv83 | 32 | 128 | 0.58 | 0.05 | 0.74 | 0.05 | 0.74 | 0.05 | 5.69 | 46.19 |
| Conv84 | 32 | 288 | 0.62 | 0.05 | 0.83 | 0.05 | 0.83 | 0.05 | 3.13 | 46.19 |
| Conv85 | 128 | 32 | 0.70 | 0.05 | 0.74 | 0.05 | 0.74 | 0.05 | 26.25 | 155.4 |
| Conv86 | 32 | 128 | 0.85 | 0.05 | 0.74 | 0.05 | 0.74 | 0.05 | 3.26 | 46.19 |
| Conv87 | 32 | 288 | 0.86 | 0.05 | 0.83 | 0.05 | 0.83 | 0.05 | 1.88 | 46.19 |

| | | | | | | | | | | |
|---|---|---|---|---|---|---|---|---|---|---|
| Conv88 | 128 | 32 | 0.84 | 0.05 | 0.74 | 0.05 | 0.74 | 0.05 | 11.77 | 155.4 |
| Conv89 | 32 | 128 | 0.81 | 0.05 | 0.74 | 0.05 | 0.74 | 0.05 | 3.42 | 46.19 |
| Conv90 | 32 | 288 | 0.81 | 0.05 | 0.83 | 0.05 | 0.83 | 0.05 | 2.45 | 46.19 |
| Conv91 | 128 | 32 | 0.81 | 0.05 | 0.74 | 0.05 | 0.74 | 0.05 | 16.74 | 155.4 |
| Conv92 | 32 | 128 | 0.68 | 0.05 | 0.74 | 0.05 | 0.74 | 0.05 | 4.9 | 46.19 |
| Conv93 | 32 | 288 | 0.68 | 0.05 | 0.83 | 0.05 | 0.83 | 0.05 | 1.94 | 46.19 |
| Conv94 | 128 | 32 | 0.62 | 0.05 | 0.74 | 0.05 | 0.74 | 0.05 | 28.67 | 155.4 |
| Conv95 | 32 | 128 | 0.39 | 0.05 | 0.74 | 0.05 | 0.74 | 0.05 | 7.35 | 46.19 |
| Conv96 | 32 | 288 | 0.62 | 0.05 | 0.83 | 0.05 | 0.83 | 0.05 | 1.82 | 46.19 |
| Conv97 | 128 | 32 | 0.67 | 0.05 | 0.74 | 0.05 | 0.74 | 0.05 | 29.33 | 155.4 |
| Conv98 | 32 | 128 | 0.88 | 0.05 | 0.74 | 0.05 | 0.74 | 0.05 | 2.17 | 46.19 |
| Conv99 | 32 | 288 | 0.88 | 0.05 | 0.83 | 0.05 | 0.83 | 0.05 | 1.18 | 46.19 |
| Conv100 | 128 | 32 | 0.87 | 0.05 | 0.74 | 0.05 | 0.74 | 0.05 | 10.3 | 155.4 |
| Conv101 | 32 | 128 | 0.71 | 0.05 | 0.74 | 0.05 | 0.74 | 0.05 | 5.73 | 46.19 |
| Conv102 | 32 | 288 | 0.67 | 0.05 | 0.83 | 0.05 | 0.83 | 0.05 | 3.25 | 46.19 |
| Conv103 | 128 | 32 | 0.68 | 0.05 | 0.74 | 0.05 | 0.74 | 0.05 | 24.74 | 155.4 |
| Conv104 | 32 | 128 | 0.81 | 0.05 | 0.74 | 0.05 | 0.74 | 0.05 | 3.97 | 46.19 |
| Conv105 | 32 | 288 | 0.68 | 0.05 | 0.83 | 0.05 | 0.83 | 0.05 | 2.33 | 46.19 |
| Conv106 | 128 | 32 | 0.84 | 0.05 | 0.74 | 0.05 | 0.74 | 0.05 | 23.88 | 155.4 |
| Conv107 | 32 | 128 | 0.66 | 0.05 | 0.74 | 0.05 | 0.74 | 0.05 | 2.85 | 46.19 |
| Conv108 | 32 | 288 | 0.85 | 0.05 | 0.83 | 0.05 | 0.83 | 0.05 | 0.93 | 46.19 |
| Conv109 | 128 | 32 | 0.86 | 0.05 | 0.74 | 0.05 | 0.74 | 0.05 | 17.92 | 155.4 |
| Conv110 | 64 | 128 | 0.54 | 0.05 | 0.82 | 0.05 | 0.82 | 0.05 | 9.86 | 83.68 |
| Conv111 | 64 | 576 | 0.69 | 0.05 | 0.97 | 0.05 | 0.97 | 0.05 | 3.44 | 83.68 |
| Conv112 | 256 | 64 | 0.53 | 0.05 | 0.90 | 0.05 | 0.90 | 0.05 | 72.59 | 294.32 |
| Conv113 | 64 | 256 | 0.81 | 0.05 | 0.90 | 0.05 | 0.90 | 0.05 | 2.17 | 83.68 |
| Conv114 | 64 | 576 | 0.80 | 0.05 | 0.97 | 0.05 | 0.97 | 0.05 | 4.2 | 83.68 |
| Conv115 | 256 | 64 | 0.60 | 0.05 | 0.90 | 0.05 | 0.90 | 0.05 | 27.87 | 294.32 |
| Conv116 | 64 | 256 | 0.82 | 0.05 | 0.90 | 0.05 | 0.90 | 0.05 | 2.3 | 83.68 |
| Conv117 | 64 | 576 | 0.84 | 0.05 | 0.97 | 0.05 | 0.97 | 0.05 | 1.58 | 83.68 |
| Conv118 | 256 | 64 | 0.83 | 0.05 | 0.90 | 0.05 | 0.90 | 0.05 | 14.32 | 294.32 |
| Conv119 | 64 | 256 | 0.82 | 0.05 | 0.90 | 0.05 | 0.90 | 0.05 | 2.93 | 83.68 |
| Conv120 | 64 | 576 | 0.70 | 0.05 | 0.97 | 0.05 | 0.97 | 0.05 | 1.69 | 83.68 |
| Conv121 | 256 | 64 | 0.80 | 0.05 | 0.90 | 0.05 | 0.90 | 0.05 | 12.83 | 294.32 |
| Conv122 | 64 | 256 | 0.68 | 0.05 | 0.90 | 0.05 | 0.90 | 0.05 | 2.67 | 83.68 |
| Conv123 | 64 | 576 | 0.65 | 0.05 | 0.97 | 0.05 | 0.97 | 0.05 | 2.46 | 83.68 |
| Conv124 | 256 | 64 | 0.70 | 0.05 | 0.90 | 0.05 | 0.90 | 0.05 | 16.62 | 294.32 |
| Conv125 | 64 | 256 | 0.70 | 0.05 | 0.90 | 0.05 | 0.90 | 0.05 | 2.21 | 83.68 |
| Conv126 | 64 | 576 | 0.81 | 0.05 | 0.97 | 0.05 | 0.97 | 0.05 | 1.82 | 83.68 |
| Conv127 | 256 | 64 | 0.67 | 0.05 | 0.90 | 0.05 | 0.90 | 0.05 | 17.12 | 294.32 |
| Conv128 | 64 | 256 | 0.61 | 0.05 | 0.90 | 0.05 | 0.90 | 0.05 | 2.58 | 83.68 |
| Conv129 | 64 | 576 | 0.70 | 0.05 | 0.97 | 0.05 | 0.97 | 0.05 | 1.92 | 83.68 |
| Conv130 | 256 | 64 | 0.58 | 0.05 | 0.90 | 0.05 | 0.90 | 0.05 | 18.45 | 294.32 |
| Conv131 | 64 | 256 | 0.63 | 0.05 | 0.90 | 0.05 | 0.90 | 0.05 | 2.37 | 83.68 |
| Conv132 | 64 | 576 | 0.81 | 0.05 | 0.97 | 0.05 | 0.97 | 0.05 | 1.24 | 83.68 |
| Conv133 | 256 | 64 | 0.70 | 0.05 | 0.90 | 0.05 | 0.90 | 0.05 | 21.78 | 294.32 |
| Conv134 | 64 | 256 | 0.39 | 0.05 | 0.90 | 0.05 | 0.90 | 0.05 | 3.23 | 83.68 |
| Conv135 | 64 | 576 | 0.71 | 0.05 | 0.97 | 0.05 | 0.97 | 0.05 | 1.68 | 83.68 |
| Conv136 | 256 | 64 | 0.60 | 0.05 | 0.90 | 0.05 | 0.90 | 0.05 | 25.14 | 294.32 |
| Conv137 | 64 | 256 | 0.59 | 0.05 | 0.90 | 0.05 | 0.90 | 0.05 | 1.62 | 83.68 |
| Conv138 | 64 | 576 | 0.84 | 0.05 | 0.97 | 0.05 | 0.97 | 0.05 | 0.9 | 83.68 |
| Conv139 | 256 | 64 | 0.58 | 0.05 | 0.90 | 0.05 | 0.90 | 0.05 | 23.89 | 294.32 |
| Conv140 | 64 | 256 | 0.43 | 0.05 | 0.90 | 0.05 | 0.90 | 0.05 | 1.81 | 83.68 |
| Conv141 | 64 | 576 | 0.84 | 0.05 | 0.97 | 0.05 | 0.97 | 0.05 | 1.01 | 83.68 |
| Conv142 | 256 | 64 | 0.62 | 0.05 | 0.90 | 0.05 | 0.90 | 0.05 | 23.72 | 294.32 |
| Conv143 | 64 | 256 | 0.61 | 0.05 | 0.90 | 0.05 | 0.90 | 0.05 | 1.68 | 83.68 |
| Conv144 | 64 | 576 | 0.83 | 0.05 | 0.97 | 0.05 | 0.97 | 0.05 | 0.68 | 83.68 |
| Conv145 | 256 | 64 | 0.62 | 0.05 | 0.90 | 0.05 | 0.90 | 0.05 | 26.18 | 294.32 |
| Conv146 | 64 | 256 | 0.62 | 0.05 | 0.90 | 0.05 | 0.90 | 0.05 | 1.48 | 83.68 |

| Layer | Number | dim | p-value | c-value | p-value | c-value | p-value | c-value | t-value | c-value |
|---|---|---|---|---|---|---|---|---|---|---|
| Conv147 | 64 | 576 | 0.83 | 0.05 | 0.97 | 0.05 | 0.97 | 0.05 | 0.64 | 83.68 |
| Conv148 | 256 | 64 | 0.59 | 0.05 | 0.90 | 0.05 | 0.90 | 0.05 | 26.75 | 294.32 |
| Conv149 | 64 | 256 | 0.66 | 0.05 | 0.90 | 0.05 | 0.90 | 0.05 | 1.97 | 83.68 |
| Conv150 | 64 | 576 | 0.84 | 0.05 | 0.97 | 0.05 | 0.97 | 0.05 | 1.09 | 83.68 |
| Conv151 | 256 | 64 | 0.59 | 0.05 | 0.90 | 0.05 | 0.90 | 0.05 | 27.51 | 294.32 |
| Conv152 | 64 | 256 | 0.48 | 0.05 | 0.90 | 0.05 | 0.90 | 0.05 | 2.36 | 83.68 |
| Conv153 | 64 | 576 | 0.87 | 0.05 | 0.97 | 0.05 | 0.97 | 0.05 | 0.58 | 83.68 |
| Conv154 | 256 | 64 | 0.57 | 0.05 | 0.90 | 0.05 | 0.90 | 0.05 | 35.04 | 294.32 |
| Conv155 | 64 | 256 | 0.58 | 0.05 | 0.90 | 0.05 | 0.90 | 0.05 | 1.68 | 83.68 |
| Conv156 | 64 | 576 | 0.85 | 0.05 | 0.97 | 0.05 | 0.97 | 0.05 | 0.6 | 83.68 |
| Conv157 | 256 | 64 | 0.53 | 0.05 | 0.90 | 0.05 | 0.90 | 0.05 | 34.74 | 294.32 |
| Conv158 | 64 | 256 | 0.48 | 0.05 | 0.90 | 0.05 | 0.90 | 0.05 | 2.37 | 83.68 |
| Conv159 | 64 | 576 | 0.85 | 0.05 | 0.97 | 0.05 | 0.97 | 0.05 | 0.8 | 83.68 |
| Conv160 | 256 | 64 | 0.53 | 0.05 | 0.90 | 0.05 | 0.90 | 0.05 | 37.4 | 294.32 |
| Conv161 | 64 | 256 | 0.47 | 0.05 | 0.90 | 0.05 | 0.90 | 0.05 | 1.81 | 83.68 |
| Conv162 | 64 | 576 | 0.85 | 0.05 | 0.97 | 0.05 | 0.97 | 0.05 | 0.58 | 83.68 |
| Conv163 | 256 | 64 | 0.49 | 0.05 | 0.90 | 0.05 | 0.90 | 0.05 | 36.85 | 294.32 |
| **Passing rate** | - | - | 97.55% | | 100.0% | | 100.0% | | 96.32% | |

## P.6 DATASET

config:

`https://github.com/bearpaw/pytorch-classification.`

Table 36: Cifar10 WRN28-10

| Layer | Number | dim | Gaussian | | Mean_Left | | Mean_Right | | Sigma | |
|---|---|---|---|---|---|---|---|---|---|---|
| | | | p-value | c-value | p-value | c-value | p-value | c-value | t-value | c-value |
| Conv1 | 16 | 27 | 0.46 | 0.05 | 0.58 | 0.05 | 0.58 | 0.05 | 27.8 | 26.3 |
| Conv2 | 160 | 144 | 0.92 | 0.05 | 0.94 | 0.05 | 0.94 | 0.05 | 4.77 | 190.52 |
| Conv3 | 160 | 1440 | 0.95 | 0.05 | 1.00 | 0.05 | 1.00 | 0.05 | 0.09 | 190.52 |
| Conv4 | 160 | 16 | 0.64 | 0.05 | 0.69 | 0.05 | 0.69 | 0.05 | 29.65 | 190.52 |
| Conv5 | 160 | 1440 | 0.94 | 0.05 | 1.00 | 0.05 | 1.00 | 0.05 | 0.5 | 190.52 |
| Conv6 | 160 | 1440 | 0.94 | 0.05 | 1.00 | 0.05 | 1.00 | 0.05 | 0.15 | 190.52 |
| Conv7 | 160 | 1440 | 0.95 | 0.05 | 1.00 | 0.05 | 1.00 | 0.05 | 0.05 | 190.52 |
| Conv8 | 160 | 1440 | 0.94 | 0.05 | 1.00 | 0.05 | 1.00 | 0.05 | 0.29 | 190.52 |
| Conv9 | 160 | 1440 | 0.94 | 0.05 | 1.00 | 0.05 | 1.00 | 0.05 | 0.08 | 190.52 |
| Conv10 | 160 | 1440 | 0.95 | 0.05 | 1.00 | 0.05 | 1.00 | 0.05 | 0.42 | 190.52 |
| Conv11 | 320 | 1440 | 0.94 | 0.05 | 1.00 | 0.05 | 1.00 | 0.05 | 1.28 | 362.72 |
| Conv12 | 320 | 2880 | 0.95 | 0.05 | 1.00 | 0.05 | 1.00 | 0.05 | 0.09 | 362.72 |
| Conv13 | 320 | 160 | 0.90 | 0.05 | 0.99 | 0.05 | 0.99 | 0.05 | 3.15 | 362.72 |
| Conv14 | 320 | 2880 | 0.95 | 0.05 | 1.00 | 0.05 | 1.00 | 0.05 | 0.12 | 362.72 |
| Conv15 | 320 | 2880 | 0.95 | 0.05 | 1.00 | 0.05 | 1.00 | 0.05 | 0.06 | 362.72 |
| Conv16 | 320 | 2880 | 0.94 | 0.05 | 1.00 | 0.05 | 1.00 | 0.05 | 0.03 | 362.72 |
| Conv17 | 320 | 2880 | 0.95 | 0.05 | 1.00 | 0.05 | 1.00 | 0.05 | 0.11 | 362.72 |
| Conv18 | 320 | 2880 | 0.94 | 0.05 | 1.00 | 0.05 | 1.00 | 0.05 | 0.04 | 362.72 |
| Conv19 | 320 | 2880 | 0.94 | 0.05 | 1.00 | 0.05 | 1.00 | 0.05 | 0.26 | 362.72 |
| Conv20 | 640 | 2880 | 0.94 | 0.05 | 1.00 | 0.05 | 1.00 | 0.05 | 3.11 | 699.96 |
| Conv21 | 640 | 5760 | 0.95 | 0.05 | 1.00 | 0.05 | 1.00 | 0.05 | 0.21 | 699.96 |
| Conv22 | 640 | 320 | 0.90 | 0.05 | 1.00 | 0.05 | 1.00 | 0.05 | 1.93 | 699.96 |
| Conv23 | 640 | 5760 | 0.95 | 0.05 | 1.00 | 0.05 | 1.00 | 0.05 | 0.91 | 699.96 |
| Conv24 | 640 | 5760 | 0.94 | 0.05 | 1.00 | 0.05 | 1.00 | 0.05 | 0.21 | 699.96 |
| Conv25 | 640 | 5760 | 0.95 | 0.05 | 1.00 | 0.05 | 1.00 | 0.05 | 0.16 | 699.96 |
| Conv26 | 640 | 5760 | 0.95 | 0.05 | 1.00 | 0.05 | 1.00 | 0.05 | 0.37 | 699.96 |
| Conv27 | 640 | 5760 | 0.95 | 0.05 | 1.00 | 0.05 | 1.00 | 0.05 | 0.05 | 699.96 |

| Conv28 | 640 | 5760 | 0.95 | 0.05 | 1.00 | 0.05 | 1.00 | 0.05 | 1.32 | 699.96 |
| **Passing rate** | - | - | 100.0% | | 100.0% | | 100.0% | | 96.43% | |

Table 37: ImageNet WRN28-10

| Layer | Number | dim | Gaussian | | Mean_Left | | Mean_Right | | Sigma | |
|---|---|---|---|---|---|---|---|---|---|---|
| | | | **p-value** | **c-value** | **p-value** | **c-value** | **p-value** | **c-value** | **t-value** | **c-value** |
| Conv1 | 16 | 27 | 0.40 | 0.05 | 0.58 | 0.05 | 0.58 | 0.05 | 30.53 | 26.3 |
| Conv2 | 160 | 144 | 0.91 | 0.05 | 0.94 | 0.05 | 0.94 | 0.05 | 6.08 | 190.52 |
| Conv3 | 160 | 1440 | 0.95 | 0.05 | 1.00 | 0.05 | 1.00 | 0.05 | 0.13 | 190.52 |
| Conv4 | 160 | 16 | 0.68 | 0.05 | 0.69 | 0.05 | 0.69 | 0.05 | 25.89 | 190.52 |
| Conv5 | 160 | 1440 | 0.94 | 0.05 | 1.00 | 0.05 | 1.00 | 0.05 | 0.34 | 190.52 |
| Conv6 | 160 | 1440 | 0.95 | 0.05 | 1.00 | 0.05 | 1.00 | 0.05 | 0.19 | 190.52 |
| Conv7 | 160 | 1440 | 0.94 | 0.05 | 1.00 | 0.05 | 1.00 | 0.05 | 0.14 | 190.52 |
| Conv8 | 160 | 1440 | 0.94 | 0.05 | 1.00 | 0.05 | 1.00 | 0.05 | 0.28 | 190.52 |
| Conv9 | 160 | 1440 | 0.94 | 0.05 | 1.00 | 0.05 | 1.00 | 0.05 | 0.03 | 190.52 |
| Conv10 | 160 | 1440 | 0.95 | 0.05 | 1.00 | 0.05 | 1.00 | 0.05 | 0.36 | 190.52 |
| Conv11 | 320 | 1440 | 0.95 | 0.05 | 1.00 | 0.05 | 1.00 | 0.05 | 1.54 | 362.72 |
| Conv12 | 320 | 2880 | 0.95 | 0.05 | 1.00 | 0.05 | 1.00 | 0.05 | 0.06 | 362.72 |
| Conv13 | 320 | 160 | 0.89 | 0.05 | 0.99 | 0.05 | 0.99 | 0.05 | 3.39 | 362.72 |
| Conv14 | 320 | 2880 | 0.94 | 0.05 | 1.00 | 0.05 | 1.00 | 0.05 | 0.11 | 362.72 |
| Conv15 | 320 | 2880 | 0.95 | 0.05 | 1.00 | 0.05 | 1.00 | 0.05 | 0.06 | 362.72 |
| Conv16 | 320 | 2880 | 0.95 | 0.05 | 1.00 | 0.05 | 1.00 | 0.05 | 0.04 | 362.72 |
| Conv17 | 320 | 2880 | 0.95 | 0.05 | 1.00 | 0.05 | 1.00 | 0.05 | 0.11 | 362.72 |
| Conv18 | 320 | 2880 | 0.95 | 0.05 | 1.00 | 0.05 | 1.00 | 0.05 | 0.03 | 362.72 |
| Conv19 | 320 | 2880 | 0.95 | 0.05 | 1.00 | 0.05 | 1.00 | 0.05 | 0.5 | 362.72 |
| Conv20 | 640 | 2880 | 0.94 | 0.05 | 1.00 | 0.05 | 1.00 | 0.05 | 3.36 | 699.96 |
| Conv21 | 640 | 5760 | 0.94 | 0.05 | 1.00 | 0.05 | 1.00 | 0.05 | 0.17 | 699.96 |
| Conv22 | 640 | 320 | 0.90 | 0.05 | 1.00 | 0.05 | 1.00 | 0.05 | 1.98 | 699.96 |
| Conv23 | 640 | 5760 | 0.94 | 0.05 | 1.00 | 0.05 | 1.00 | 0.05 | 0.74 | 699.96 |
| Conv24 | 640 | 5760 | 0.95 | 0.05 | 1.00 | 0.05 | 1.00 | 0.05 | 0.2 | 699.96 |
| Conv25 | 640 | 5760 | 0.95 | 0.05 | 1.00 | 0.05 | 1.00 | 0.05 | 0.23 | 699.96 |
| Conv26 | 640 | 5760 | 0.94 | 0.05 | 1.00 | 0.05 | 1.00 | 0.05 | 0.36 | 699.96 |
| Conv27 | 640 | 5760 | 0.95 | 0.05 | 1.00 | 0.05 | 1.00 | 0.05 | 0.05 | 699.96 |
| Conv28 | 640 | 5760 | 0.94 | 0.05 | 1.00 | 0.05 | 1.00 | 0.05 | 1.3 | 699.96 |
| **Passing rate** | - | - | 100.0% | | 100.0% | | 100.0% | | 96.43% | |

Table 38: MNIST WRN28-10

| Layer | Number | dim | Gaussian | | Mean_Left | | Mean_Right | | Sigma | |
|---|---|---|---|---|---|---|---|---|---|---|
| | | | **p-value** | **c-value** | **p-value** | **c-value** | **p-value** | **c-value** | **t-value** | **c-value** |
| Conv1 | 16 | 27 | 0.46 | 0.05 | 0.58 | 0.05 | 0.58 | 0.05 | 33.06 | 26.3 |
| Conv2 | 160 | 144 | 0.92 | 0.05 | 0.94 | 0.05 | 0.94 | 0.05 | 5.57 | 190.52 |
| Conv3 | 160 | 1440 | 0.95 | 0.05 | 1.00 | 0.05 | 1.00 | 0.05 | 0.1 | 190.52 |
| Conv4 | 160 | 16 | 0.64 | 0.05 | 0.69 | 0.05 | 0.69 | 0.05 | 30.62 | 190.52 |
| Conv5 | 160 | 1440 | 0.95 | 0.05 | 1.00 | 0.05 | 1.00 | 0.05 | 0.5 | 190.52 |
| Conv6 | 160 | 1440 | 0.95 | 0.05 | 1.00 | 0.05 | 1.00 | 0.05 | 0.13 | 190.52 |
| Conv7 | 160 | 1440 | 0.94 | 0.05 | 1.00 | 0.05 | 1.00 | 0.05 | 0.1 | 190.52 |
| Conv8 | 160 | 1440 | 0.95 | 0.05 | 1.00 | 0.05 | 1.00 | 0.05 | 0.28 | 190.52 |
| Conv9 | 160 | 1440 | 0.95 | 0.05 | 1.00 | 0.05 | 1.00 | 0.05 | 0.02 | 190.52 |
| Conv10 | 160 | 1440 | 0.95 | 0.05 | 1.00 | 0.05 | 1.00 | 0.05 | 0.43 | 190.52 |

| Layer | Number | dim | | | | | | | | |
|-------|--------|-----|------|------|------|------|------|------|------|--------|
| Conv11 | 320 | 1440 | 0.94 | 0.05 | 1.00 | 0.05 | 1.00 | 0.05 | 1.56 | 362.72 |
| Conv12 | 320 | 2880 | 0.95 | 0.05 | 1.00 | 0.05 | 1.00 | 0.05 | 0.08 | 362.72 |
| Conv13 | 320 | 160 | 0.90 | 0.05 | 0.99 | 0.05 | 0.99 | 0.05 | 3.91 | 362.72 |
| Conv14 | 320 | 2880 | 0.95 | 0.05 | 1.00 | 0.05 | 1.00 | 0.05 | 0.17 | 362.72 |
| Conv15 | 320 | 2880 | 0.95 | 0.05 | 1.00 | 0.05 | 1.00 | 0.05 | 0.06 | 362.72 |
| Conv16 | 320 | 2880 | 0.94 | 0.05 | 1.00 | 0.05 | 1.00 | 0.05 | 0.03 | 362.72 |
| Conv17 | 320 | 2880 | 0.95 | 0.05 | 1.00 | 0.05 | 1.00 | 0.05 | 0.09 | 362.72 |
| Conv18 | 320 | 2880 | 0.95 | 0.05 | 1.00 | 0.05 | 1.00 | 0.05 | 0.03 | 362.72 |
| Conv19 | 320 | 2880 | 0.95 | 0.05 | 1.00 | 0.05 | 1.00 | 0.05 | 0.5 | 362.72 |
| Conv20 | 640 | 2880 | 0.94 | 0.05 | 1.00 | 0.05 | 1.00 | 0.05 | 2.85 | 699.96 |
| Conv21 | 640 | 5760 | 0.94 | 0.05 | 1.00 | 0.05 | 1.00 | 0.05 | 0.14 | 699.96 |
| Conv22 | 640 | 320 | 0.89 | 0.05 | 1.00 | 0.05 | 1.00 | 0.05 | 2.34 | 699.96 |
| Conv23 | 640 | 5760 | 0.94 | 0.05 | 1.00 | 0.05 | 1.00 | 0.05 | 0.68 | 699.96 |
| Conv24 | 640 | 5760 | 0.95 | 0.05 | 1.00 | 0.05 | 1.00 | 0.05 | 0.23 | 699.96 |
| Conv25 | 640 | 5760 | 0.95 | 0.05 | 1.00 | 0.05 | 1.00 | 0.05 | 0.19 | 699.96 |
| Conv26 | 640 | 5760 | 0.95 | 0.05 | 1.00 | 0.05 | 1.00 | 0.05 | 0.44 | 699.96 |
| Conv27 | 640 | 5760 | 0.95 | 0.05 | 1.00 | 0.05 | 1.00 | 0.05 | 0.05 | 699.96 |
| Conv28 | 640 | 5760 | 0.94 | 0.05 | 1.00 | 0.05 | 1.00 | 0.05 | 1.5 | 699.96 |
| **Passing rate** | - | - | 100.0% | | 100.0% | | 100.0% | | 96.43% | |

## P.7 OTHER TASKS

### P.7.1 SEGMENTATION

config:

`https://github.com/meetshah1995/pytorch-semse`

`https://github.com/speedinghzl/pytorch-segmentation-toolbox`

Table 39: Cityscapes SegNet

| Layer | Number | dim | Gaussian | | Mean_Left | | Mean_Right | | Sigma | |
|-------|--------|-----|----------|----------|-----------|----------|------------|----------|---------|---------|
| | | | p-value | c-value | p-value | c-value | p-value | c-value | t-value | c-value |
| Conv1 | 64 | 27 | 0.86 | 0.05 | 0.66 | 0.05 | 0.66 | 0.05 | 20.77 | 83.68 |
| Conv2 | 64 | 576 | 0.88 | 0.05 | 0.97 | 0.05 | 0.97 | 0.05 | 0.51 | 83.68 |
| Conv3 | 128 | 576 | 0.84 | 0.05 | 1.00 | 0.05 | 1.00 | 0.05 | 0.8 | 155.4 |
| Conv4 | 128 | 1152 | 0.92 | 0.05 | 1.00 | 0.05 | 1.00 | 0.05 | 0.42 | 155.4 |
| Conv5 | 256 | 1152 | 0.91 | 0.05 | 1.00 | 0.05 | 1.00 | 0.05 | 1.05 | 294.32 |
| Conv6 | 256 | 2304 | 0.93 | 0.05 | 1.00 | 0.05 | 1.00 | 0.05 | 0.48 | 294.32 |
| Conv7 | 256 | 2304 | 0.94 | 0.05 | 1.00 | 0.05 | 1.00 | 0.05 | 0.44 | 294.32 |
| Conv8 | 512 | 2304 | 0.94 | 0.05 | 1.00 | 0.05 | 1.00 | 0.05 | 0.96 | 565.75 |
| Conv9 | 512 | 4608 | 0.95 | 0.05 | 1.00 | 0.05 | 1.00 | 0.05 | 0.39 | 565.75 |
| Conv10 | 512 | 4608 | 0.95 | 0.05 | 1.00 | 0.05 | 1.00 | 0.05 | 0.47 | 565.75 |
| Conv11 | 512 | 4608 | 0.94 | 0.05 | 1.00 | 0.05 | 1.00 | 0.05 | 0.48 | 565.75 |
| Conv12 | 512 | 4608 | 0.95 | 0.05 | 1.00 | 0.05 | 1.00 | 0.05 | 0.38 | 565.75 |
| Conv13 | 512 | 4608 | 0.95 | 0.05 | 1.00 | 0.05 | 1.00 | 0.05 | 0.45 | 565.75 |
| Conv14 | 512 | 4608 | 0.95 | 0.05 | 1.00 | 0.05 | 1.00 | 0.05 | 0.36 | 565.75 |
| Conv15 | 512 | 4608 | 0.93 | 0.05 | 1.00 | 0.05 | 1.00 | 0.05 | 0.41 | 565.75 |
| Conv16 | 512 | 4608 | 0.94 | 0.05 | 1.00 | 0.05 | 1.00 | 0.05 | 0.26 | 565.75 |
| Conv17 | 512 | 4608 | 0.94 | 0.05 | 1.00 | 0.05 | 1.00 | 0.05 | 0.57 | 565.75 |
| Conv18 | 512 | 4608 | 0.94 | 0.05 | 1.00 | 0.05 | 1.00 | 0.05 | 0.32 | 565.75 |
| Conv19 | 256 | 4608 | 0.93 | 0.05 | 1.00 | 0.05 | 1.00 | 0.05 | 0.14 | 294.32 |
| Conv20 | 256 | 2304 | 0.95 | 0.05 | 1.00 | 0.05 | 1.00 | 0.05 | 0.42 | 294.32 |
| Conv21 | 256 | 2304 | 0.95 | 0.05 | 1.00 | 0.05 | 1.00 | 0.05 | 0.28 | 294.32 |
| Conv22 | 128 | 2304 | 0.94 | 0.05 | 1.00 | 0.05 | 1.00 | 0.05 | 0.17 | 155.4 |
| Conv23 | 128 | 1152 | 0.91 | 0.05 | 1.00 | 0.05 | 1.00 | 0.05 | 0.29 | 155.4 |

| Layer | Number | dim | | | | | | | | |
|---|---|---|---|---|---|---|---|---|---|---|
| Conv24 | 64 | 1152 | 0.93 | 0.05 | 1.00 | 0.05 | 1.00 | 0.05 | 0.15 | 83.68 |
| Conv25 | 64 | 576 | 0.92 | 0.05 | 0.97 | 0.05 | 0.97 | 0.05 | 0.31 | 83.68 |
| Conv26 | 19 | 576 | 0.87 | 0.05 | 0.85 | 0.05 | 0.85 | 0.05 | 0.9 | 30.14 |
| **Passing rate** | - | - | 100.0% | | 100.0% | | 100.0% | | 100.0% | |

Table 40: Cityscapes PSPNet

| Layer | Number | dim | Gaussian | | Mean_Left | | Mean_Right | | Sigma | |
|---|---|---|---|---|---|---|---|---|---|---|
| | | | p-value | c-value | p-value | c-value | p-value | c-value | t-value | c-value |
| Conv1 | 64 | 27 | 0.38 | 0.05 | 0.66 | 0.05 | 0.66 | 0.05 | 286.98 | 83.68 |
| Conv2 | 64 | 576 | 0.49 | 0.05 | 0.97 | 0.05 | 0.97 | 0.05 | 12.81 | 83.68 |
| Conv3 | 128 | 576 | 0.70 | 0.05 | 1.00 | 0.05 | 1.00 | 0.05 | 1.47 | 155.4 |
| Conv4 | 64 | 128 | 0.85 | 0.05 | 0.82 | 0.05 | 0.82 | 0.05 | 4.77 | 83.68 |
| Conv5 | 64 | 576 | 0.87 | 0.05 | 0.97 | 0.05 | 0.97 | 0.05 | 0.95 | 83.68 |
| Conv6 | 256 | 64 | 0.85 | 0.05 | 0.90 | 0.05 | 0.90 | 0.05 | 15.16 | 294.32 |
| Conv7 | 256 | 128 | 0.60 | 0.05 | 0.96 | 0.05 | 0.96 | 0.05 | 17.95 | 294.32 |
| Conv8 | 64 | 256 | 0.91 | 0.05 | 0.90 | 0.05 | 0.90 | 0.05 | 1.78 | 83.68 |
| Conv9 | 64 | 576 | 0.87 | 0.05 | 0.97 | 0.05 | 0.97 | 0.05 | 0.25 | 83.68 |
| Conv10 | 256 | 64 | 0.88 | 0.05 | 0.90 | 0.05 | 0.90 | 0.05 | 10.76 | 294.32 |
| Conv11 | 64 | 256 | 0.91 | 0.05 | 0.90 | 0.05 | 0.90 | 0.05 | 2.01 | 83.68 |
| Conv12 | 64 | 576 | 0.91 | 0.05 | 0.97 | 0.05 | 0.97 | 0.05 | 0.35 | 83.68 |
| Conv13 | 256 | 64 | 0.89 | 0.05 | 0.90 | 0.05 | 0.90 | 0.05 | 10.08 | 294.32 |
| Conv14 | 128 | 256 | 0.86 | 0.05 | 0.96 | 0.05 | 0.96 | 0.05 | 1.76 | 155.4 |
| Conv15 | 128 | 1152 | 0.92 | 0.05 | 1.00 | 0.05 | 1.00 | 0.05 | 0.07 | 155.4 |
| Conv16 | 512 | 128 | 0.87 | 0.05 | 0.99 | 0.05 | 0.99 | 0.05 | 21.04 | 565.75 |
| Conv17 | 512 | 256 | 0.85 | 0.05 | 1.00 | 0.05 | 1.00 | 0.05 | 11.71 | 565.75 |
| Conv18 | 128 | 512 | 0.91 | 0.05 | 0.99 | 0.05 | 0.99 | 0.05 | 0.45 | 155.4 |
| Conv19 | 128 | 1152 | 0.89 | 0.05 | 1.00 | 0.05 | 1.00 | 0.05 | 1.22 | 155.4 |
| Conv20 | 512 | 128 | 0.90 | 0.05 | 0.99 | 0.05 | 0.99 | 0.05 | 12.48 | 565.75 |
| Conv21 | 128 | 512 | 0.93 | 0.05 | 0.99 | 0.05 | 0.99 | 0.05 | 0.22 | 155.4 |
| Conv22 | 128 | 1152 | 0.90 | 0.05 | 1.00 | 0.05 | 1.00 | 0.05 | 0.17 | 155.4 |
| Conv23 | 512 | 128 | 0.87 | 0.05 | 0.99 | 0.05 | 0.99 | 0.05 | 8.64 | 565.75 |
| Conv24 | 128 | 512 | 0.91 | 0.05 | 0.99 | 0.05 | 0.99 | 0.05 | 0.12 | 155.4 |
| Conv25 | 128 | 1152 | 0.92 | 0.05 | 1.00 | 0.05 | 1.00 | 0.05 | 0.11 | 155.4 |
| Conv26 | 512 | 128 | 0.89 | 0.05 | 0.99 | 0.05 | 0.99 | 0.05 | 10.47 | 565.75 |
| Conv27 | 256 | 512 | 0.87 | 0.05 | 1.00 | 0.05 | 1.00 | 0.05 | 0.65 | 294.32 |
| Conv28 | 256 | 2304 | 0.92 | 0.05 | 1.00 | 0.05 | 1.00 | 0.05 | 0.2 | 294.32 |
| Conv29 | 1024 | 256 | 0.89 | 0.05 | 1.00 | 0.05 | 1.00 | 0.05 | 16.07 | 1099.56 |
| Conv30 | 1024 | 512 | 0.84 | 0.05 | 1.00 | 0.05 | 1.00 | 0.05 | 10.07 | 1099.56 |
| Conv31 | 256 | 1024 | 0.90 | 0.05 | 1.00 | 0.05 | 1.00 | 0.05 | 0.35 | 294.32 |
| Conv32 | 256 | 2304 | 0.94 | 0.05 | 1.00 | 0.05 | 1.00 | 0.05 | 0.24 | 294.32 |
| Conv33 | 1024 | 256 | 0.89 | 0.05 | 1.00 | 0.05 | 1.00 | 0.05 | 7.69 | 1099.56 |
| Conv34 | 256 | 1024 | 0.91 | 0.05 | 1.00 | 0.05 | 1.00 | 0.05 | 0.26 | 294.32 |
| Conv35 | 256 | 2304 | 0.93 | 0.05 | 1.00 | 0.05 | 1.00 | 0.05 | 0.35 | 294.32 |
| Conv36 | 1024 | 256 | 0.91 | 0.05 | 1.00 | 0.05 | 1.00 | 0.05 | 7.32 | 1099.56 |
| Conv37 | 256 | 1024 | 0.86 | 0.05 | 1.00 | 0.05 | 1.00 | 0.05 | 0.3 | 294.32 |
| Conv38 | 256 | 2304 | 0.91 | 0.05 | 1.00 | 0.05 | 1.00 | 0.05 | 0.17 | 294.32 |
| Conv39 | 1024 | 256 | 0.91 | 0.05 | 1.00 | 0.05 | 1.00 | 0.05 | 6.23 | 1099.56 |
| Conv40 | 256 | 1024 | 0.91 | 0.05 | 1.00 | 0.05 | 1.00 | 0.05 | 0.18 | 294.32 |
| Conv41 | 256 | 2304 | 0.88 | 0.05 | 1.00 | 0.05 | 1.00 | 0.05 | 0.06 | 294.32 |
| Conv42 | 1024 | 256 | 0.81 | 0.05 | 1.00 | 0.05 | 1.00 | 0.05 | 11.36 | 1099.56 |
| Conv43 | 256 | 1024 | 0.92 | 0.05 | 1.00 | 0.05 | 1.00 | 0.05 | 0.15 | 294.32 |
| Conv44 | 256 | 2304 | 0.93 | 0.05 | 1.00 | 0.05 | 1.00 | 0.05 | 0.07 | 294.32 |
| Conv45 | 1024 | 256 | 0.89 | 0.05 | 1.00 | 0.05 | 1.00 | 0.05 | 7.5 | 1099.56 |
| Conv46 | 256 | 1024 | 0.91 | 0.05 | 1.00 | 0.05 | 1.00 | 0.05 | 0.17 | 294.32 |

| | | | | | | | | | | |
|---|---|---|---|---|---|---|---|---|---|---|
| Conv47 | 256 | 2304 | 0.94 | 0.05 | 1.00 | 0.05 | 1.00 | 0.05 | 0.1 | 294.32 |
| Conv48 | 1024 | 256 | 0.92 | 0.05 | 1.00 | 0.05 | 1.00 | 0.05 | 5.1 | 1099.56 |
| Conv49 | 256 | 1024 | 0.88 | 0.05 | 1.00 | 0.05 | 1.00 | 0.05 | 0.17 | 294.32 |
| Conv50 | 256 | 2304 | 0.93 | 0.05 | 1.00 | 0.05 | 1.00 | 0.05 | 0.09 | 294.32 |
| Conv51 | 1024 | 256 | 0.88 | 0.05 | 1.00 | 0.05 | 1.00 | 0.05 | 7.45 | 1099.56 |
| Conv52 | 256 | 1024 | 0.87 | 0.05 | 1.00 | 0.05 | 1.00 | 0.05 | 0.15 | 294.32 |
| Conv53 | 256 | 2304 | 0.92 | 0.05 | 1.00 | 0.05 | 1.00 | 0.05 | 0.13 | 294.32 |
| Conv54 | 1024 | 256 | 0.88 | 0.05 | 1.00 | 0.05 | 1.00 | 0.05 | 3.5 | 1099.56 |
| Conv55 | 256 | 1024 | 0.93 | 0.05 | 1.00 | 0.05 | 1.00 | 0.05 | 0.1 | 294.32 |
| Conv56 | 256 | 2304 | 0.93 | 0.05 | 1.00 | 0.05 | 1.00 | 0.05 | 0.11 | 294.32 |
| Conv57 | 1024 | 256 | 0.93 | 0.05 | 1.00 | 0.05 | 1.00 | 0.05 | 3.86 | 1099.56 |
| Conv58 | 256 | 1024 | 0.91 | 0.05 | 1.00 | 0.05 | 1.00 | 0.05 | 0.18 | 294.32 |
| Conv59 | 256 | 2304 | 0.92 | 0.05 | 1.00 | 0.05 | 1.00 | 0.05 | 0.18 | 294.32 |
| Conv60 | 1024 | 256 | 0.92 | 0.05 | 1.00 | 0.05 | 1.00 | 0.05 | 6.96 | 1099.56 |
| Conv61 | 256 | 1024 | 0.91 | 0.05 | 1.00 | 0.05 | 1.00 | 0.05 | 0.18 | 294.32 |
| Conv62 | 256 | 2304 | 0.93 | 0.05 | 1.00 | 0.05 | 1.00 | 0.05 | 0.16 | 294.32 |
| Conv63 | 1024 | 256 | 0.89 | 0.05 | 1.00 | 0.05 | 1.00 | 0.05 | 7.38 | 1099.56 |
| Conv64 | 256 | 1024 | 0.91 | 0.05 | 1.00 | 0.05 | 1.00 | 0.05 | 0.23 | 294.32 |
| Conv65 | 256 | 2304 | 0.93 | 0.05 | 1.00 | 0.05 | 1.00 | 0.05 | 0.12 | 294.32 |
| Conv66 | 1024 | 256 | 0.92 | 0.05 | 1.00 | 0.05 | 1.00 | 0.05 | 5.27 | 1099.56 |
| Conv67 | 256 | 1024 | 0.85 | 0.05 | 1.00 | 0.05 | 1.00 | 0.05 | 0.2 | 294.32 |
| Conv68 | 256 | 2304 | 0.92 | 0.05 | 1.00 | 0.05 | 1.00 | 0.05 | 0.18 | 294.32 |
| Conv69 | 1024 | 256 | 0.90 | 0.05 | 1.00 | 0.05 | 1.00 | 0.05 | 4.64 | 1099.56 |
| Conv70 | 256 | 1024 | 0.92 | 0.05 | 1.00 | 0.05 | 1.00 | 0.05 | 0.1 | 294.32 |
| Conv71 | 256 | 2304 | 0.94 | 0.05 | 1.00 | 0.05 | 1.00 | 0.05 | 0.16 | 294.32 |
| Conv72 | 1024 | 256 | 0.91 | 0.05 | 1.00 | 0.05 | 1.00 | 0.05 | 4.37 | 1099.56 |
| Conv73 | 256 | 1024 | 0.91 | 0.05 | 1.00 | 0.05 | 1.00 | 0.05 | 0.1 | 294.32 |
| Conv74 | 256 | 2304 | 0.94 | 0.05 | 1.00 | 0.05 | 1.00 | 0.05 | 0.16 | 294.32 |
| Conv75 | 1024 | 256 | 0.92 | 0.05 | 1.00 | 0.05 | 1.00 | 0.05 | 6.24 | 1099.56 |
| Conv76 | 256 | 1024 | 0.90 | 0.05 | 1.00 | 0.05 | 1.00 | 0.05 | 0.11 | 294.32 |
| Conv77 | 256 | 2304 | 0.92 | 0.05 | 1.00 | 0.05 | 1.00 | 0.05 | 0.15 | 294.32 |
| Conv78 | 1024 | 256 | 0.92 | 0.05 | 1.00 | 0.05 | 1.00 | 0.05 | 3.44 | 1099.56 |
| Conv79 | 256 | 1024 | 0.90 | 0.05 | 1.00 | 0.05 | 1.00 | 0.05 | 0.1 | 294.32 |
| Conv80 | 256 | 2304 | 0.93 | 0.05 | 1.00 | 0.05 | 1.00 | 0.05 | 0.12 | 294.32 |
| Conv81 | 1024 | 256 | 0.91 | 0.05 | 1.00 | 0.05 | 1.00 | 0.05 | 4.25 | 1099.56 |
| Conv82 | 256 | 1024 | 0.90 | 0.05 | 1.00 | 0.05 | 1.00 | 0.05 | 0.09 | 294.32 |
| Conv83 | 256 | 2304 | 0.92 | 0.05 | 1.00 | 0.05 | 1.00 | 0.05 | 0.15 | 294.32 |
| Conv84 | 1024 | 256 | 0.89 | 0.05 | 1.00 | 0.05 | 1.00 | 0.05 | 5.13 | 1099.56 |
| Conv85 | 256 | 1024 | 0.90 | 0.05 | 1.00 | 0.05 | 1.00 | 0.05 | 0.09 | 294.32 |
| Conv86 | 256 | 2304 | 0.93 | 0.05 | 1.00 | 0.05 | 1.00 | 0.05 | 0.17 | 294.32 |
| Conv87 | 1024 | 256 | 0.88 | 0.05 | 1.00 | 0.05 | 1.00 | 0.05 | 4.76 | 1099.56 |
| Conv88 | 256 | 1024 | 0.90 | 0.05 | 1.00 | 0.05 | 1.00 | 0.05 | 0.09 | 294.32 |
| Conv89 | 256 | 2304 | 0.89 | 0.05 | 1.00 | 0.05 | 1.00 | 0.05 | 0.09 | 294.32 |
| Conv90 | 1024 | 256 | 0.89 | 0.05 | 1.00 | 0.05 | 1.00 | 0.05 | 3.86 | 1099.56 |
| Conv91 | 256 | 1024 | 0.92 | 0.05 | 1.00 | 0.05 | 1.00 | 0.05 | 0.07 | 294.32 |
| Conv92 | 256 | 2304 | 0.92 | 0.05 | 1.00 | 0.05 | 1.00 | 0.05 | 0.03 | 294.32 |
| Conv93 | 1024 | 256 | 0.92 | 0.05 | 1.00 | 0.05 | 1.00 | 0.05 | 4.57 | 1099.56 |
| Conv94 | 256 | 1024 | 0.91 | 0.05 | 1.00 | 0.05 | 1.00 | 0.05 | 0.12 | 294.32 |
| Conv95 | 256 | 2304 | 0.93 | 0.05 | 1.00 | 0.05 | 1.00 | 0.05 | 0.07 | 294.32 |
| Conv96 | 1024 | 256 | 0.89 | 0.05 | 1.00 | 0.05 | 1.00 | 0.05 | 6.76 | 1099.56 |
| Conv97 | 512 | 1024 | 0.90 | 0.05 | 1.00 | 0.05 | 1.00 | 0.05 | 0.08 | 565.75 |
| Conv98 | 512 | 4608 | 0.94 | 0.05 | 1.00 | 0.05 | 1.00 | 0.05 | 0.02 | 565.75 |
| Conv99 | 2048 | 512 | 0.89 | 0.05 | 1.00 | 0.05 | 1.00 | 0.05 | 1.67 | 2154.4 |
| Conv100 | 2048 | 1024 | 0.69 | 0.05 | 1.00 | 0.05 | 1.00 | 0.05 | 1.49 | 2154.4 |
| Conv101 | 512 | 2048 | 0.88 | 0.05 | 1.00 | 0.05 | 1.00 | 0.05 | 0.05 | 565.75 |
| Conv102 | 512 | 4608 | 0.90 | 0.05 | 1.00 | 0.05 | 1.00 | 0.05 | 0.02 | 565.75 |
| Conv103 | 2048 | 512 | 0.90 | 0.05 | 1.00 | 0.05 | 1.00 | 0.05 | 1.95 | 2154.4 |
| Conv104 | 512 | 2048 | 0.85 | 0.05 | 1.00 | 0.05 | 1.00 | 0.05 | 0.07 | 565.75 |
| Conv105 | 512 | 4608 | 0.95 | 0.05 | 1.00 | 0.05 | 1.00 | 0.05 | 0.01 | 565.75 |

| Layer | Number | dim | | | | | | | | |
|-------|--------|-----|------|------|------|------|------|------|------|--------|
| Conv106 | 2048 | 512 | 0.90 | 0.05 | 1.00 | 0.05 | 1.00 | 0.05 | 1.9 | 2154.4 |
| Conv107 | 512 | 2048 | 0.96 | 0.05 | 1.00 | 0.05 | 1.00 | 0.05 | 0.01 | 565.75 |
| Conv108 | 512 | 2048 | 0.96 | 0.05 | 1.00 | 0.05 | 1.00 | 0.05 | 0.0 | 565.75 |
| Conv109 | 512 | 2048 | 0.96 | 0.05 | 1.00 | 0.05 | 1.00 | 0.05 | 0.0 | 565.75 |
| Conv110 | 512 | 2048 | 0.96 | 0.05 | 1.00 | 0.05 | 1.00 | 0.05 | 0.0 | 565.75 |
| Conv111 | 512 | 36864 | 0.95 | 0.05 | 1.00 | 0.05 | 1.00 | 0.05 | 0.0 | 565.75 |
| Conv112 | 19 | 512 | 0.88 | 0.05 | 0.84 | 0.05 | 0.84 | 0.05 | 0.99 | 30.14 |
| Conv113 | 512 | 9216 | 0.96 | 0.05 | 1.00 | 0.05 | 1.00 | 0.05 | 0.0 | 565.75 |
| Conv114 | 19 | 512 | 0.87 | 0.05 | 0.84 | 0.05 | 0.84 | 0.05 | 1.06 | 30.14 |
| **Passing rate** | - | - | 100.0% | | 100.0% | | 100.0% | | 99.12% | |

## P.7.2 FASTER RCNN

config:

`https://github.com/jwyang/faster-rcnn.pytorch`

Table 41: COCO ResNet101 max epoch=6

| Layer | Number | dim | Gaussian | | Mean_Left | | Mean_Right | | Sigma | |
|-------|--------|-----|----------|---------|-----------|---------|------------|---------|---------|---------|
| | | | p-value | c-value | p-value | c-value | p-value | c-value | t-value | c-value |
| Conv1 | 64 | 64 | 0.57 | 0.05 | 0.74 | 0.05 | 0.74 | 0.05 | 23.34 | 83.68 |
| Conv2 | 64 | 576 | 0.69 | 0.05 | 0.97 | 0.05 | 0.97 | 0.05 | 5.71 | 83.68 |
| Conv3 | 256 | 64 | 0.80 | 0.05 | 0.90 | 0.05 | 0.90 | 0.05 | 28.22 | 294.32 |
| Conv4 | 256 | 64 | 0.46 | 0.05 | 0.90 | 0.05 | 0.90 | 0.05 | 77.88 | 294.32 |
| Conv5 | 64 | 256 | 0.86 | 0.05 | 0.90 | 0.05 | 0.90 | 0.05 | 3.87 | 83.68 |
| Conv6 | 64 | 576 | 0.67 | 0.05 | 0.97 | 0.05 | 0.97 | 0.05 | 2.11 | 83.68 |
| Conv7 | 256 | 64 | 0.85 | 0.05 | 0.90 | 0.05 | 0.90 | 0.05 | 22.48 | 294.32 |
| Conv8 | 64 | 256 | 0.85 | 0.05 | 0.90 | 0.05 | 0.90 | 0.05 | 2.53 | 83.68 |
| Conv9 | 64 | 576 | 0.90 | 0.05 | 0.97 | 0.05 | 0.97 | 0.05 | 0.13 | 83.68 |
| Conv10 | 256 | 64 | 0.87 | 0.05 | 0.90 | 0.05 | 0.90 | 0.05 | 30.9 | 294.32 |
| Conv11 | 128 | 256 | 0.62 | 0.05 | 0.96 | 0.05 | 0.96 | 0.05 | 3.99 | 155.4 |
| Conv12 | 128 | 1152 | 0.83 | 0.05 | 1.00 | 0.05 | 1.00 | 0.05 | 2.08 | 155.4 |
| Conv13 | 512 | 128 | 0.66 | 0.05 | 0.99 | 0.05 | 0.99 | 0.05 | 27.85 | 565.75 |
| Conv14 | 512 | 256 | 0.55 | 0.05 | 1.00 | 0.05 | 1.00 | 0.05 | 18.51 | 565.75 |
| Conv15 | 128 | 512 | 0.90 | 0.05 | 0.99 | 0.05 | 0.99 | 0.05 | 0.52 | 155.4 |
| Conv16 | 128 | 1152 | 0.86 | 0.05 | 1.00 | 0.05 | 1.00 | 0.05 | 0.36 | 155.4 |
| Conv17 | 512 | 128 | 0.67 | 0.05 | 0.99 | 0.05 | 0.99 | 0.05 | 15.55 | 565.75 |
| Conv18 | 128 | 512 | 0.87 | 0.05 | 0.99 | 0.05 | 0.99 | 0.05 | 0.86 | 155.4 |
| Conv19 | 128 | 1152 | 0.91 | 0.05 | 1.00 | 0.05 | 1.00 | 0.05 | 0.35 | 155.4 |
| Conv20 | 512 | 128 | 0.71 | 0.05 | 0.99 | 0.05 | 0.99 | 0.05 | 16.78 | 565.75 |
| Conv21 | 128 | 512 | 0.86 | 0.05 | 0.99 | 0.05 | 0.99 | 0.05 | 0.98 | 155.4 |
| Conv22 | 128 | 1152 | 0.89 | 0.05 | 1.00 | 0.05 | 1.00 | 0.05 | 0.27 | 155.4 |
| Conv23 | 512 | 128 | 0.83 | 0.05 | 0.99 | 0.05 | 0.99 | 0.05 | 27.72 | 565.75 |
| Conv24 | 256 | 512 | 0.87 | 0.05 | 1.00 | 0.05 | 1.00 | 0.05 | 0.85 | 294.32 |
| Conv25 | 256 | 2304 | 0.90 | 0.05 | 1.00 | 0.05 | 1.00 | 0.05 | 0.49 | 294.32 |
| Conv26 | 1024 | 256 | 0.86 | 0.05 | 1.00 | 0.05 | 1.00 | 0.05 | 12.09 | 1099.56 |
| Conv27 | 1024 | 512 | 0.69 | 0.05 | 1.00 | 0.05 | 1.00 | 0.05 | 12.68 | 1099.56 |
| Conv28 | 256 | 1024 | 0.90 | 0.05 | 1.00 | 0.05 | 1.00 | 0.05 | 0.4 | 294.32 |
| Conv29 | 256 | 2304 | 0.91 | 0.05 | 1.00 | 0.05 | 1.00 | 0.05 | 0.63 | 294.32 |
| Conv30 | 1024 | 256 | 0.86 | 0.05 | 1.00 | 0.05 | 1.00 | 0.05 | 6.72 | 1099.56 |
| Conv31 | 256 | 1024 | 0.91 | 0.05 | 1.00 | 0.05 | 1.00 | 0.05 | 0.27 | 294.32 |
| Conv32 | 256 | 2304 | 0.93 | 0.05 | 1.00 | 0.05 | 1.00 | 0.05 | 0.25 | 294.32 |
| Conv33 | 1024 | 256 | 0.90 | 0.05 | 1.00 | 0.05 | 1.00 | 0.05 | 4.71 | 1099.56 |
| Conv34 | 256 | 1024 | 0.91 | 0.05 | 1.00 | 0.05 | 1.00 | 0.05 | 0.39 | 294.32 |
| Conv35 | 256 | 2304 | 0.93 | 0.05 | 1.00 | 0.05 | 1.00 | 0.05 | 0.16 | 294.32 |

| | | | | | | | | | | |
|---|---|---|---|---|---|---|---|---|---|---|
| Conv36 | 1024 | 256 | 0.87 | 0.05 | 1.00 | 0.05 | 1.00 | 0.05 | 9.99 | 1099.56 |
| Conv37 | 256 | 1024 | 0.92 | 0.05 | 1.00 | 0.05 | 1.00 | 0.05 | 0.26 | 294.32 |
| Conv38 | 256 | 2304 | 0.93 | 0.05 | 1.00 | 0.05 | 1.00 | 0.05 | 0.2 | 294.32 |
| Conv39 | 1024 | 256 | 0.88 | 0.05 | 1.00 | 0.05 | 1.00 | 0.05 | 4.41 | 1099.56 |
| Conv40 | 256 | 1024 | 0.91 | 0.05 | 1.00 | 0.05 | 1.00 | 0.05 | 0.23 | 294.32 |
| Conv41 | 256 | 2304 | 0.93 | 0.05 | 1.00 | 0.05 | 1.00 | 0.05 | 0.24 | 294.32 |
| Conv42 | 1024 | 256 | 0.91 | 0.05 | 1.00 | 0.05 | 1.00 | 0.05 | 4.76 | 1099.56 |
| Conv43 | 256 | 1024 | 0.85 | 0.05 | 1.00 | 0.05 | 1.00 | 0.05 | 0.33 | 294.32 |
| Conv44 | 256 | 2304 | 0.92 | 0.05 | 1.00 | 0.05 | 1.00 | 0.05 | 0.31 | 294.32 |
| Conv45 | 1024 | 256 | 0.89 | 0.05 | 1.00 | 0.05 | 1.00 | 0.05 | 7.7 | 1099.56 |
| Conv46 | 256 | 1024 | 0.89 | 0.05 | 1.00 | 0.05 | 1.00 | 0.05 | 0.25 | 294.32 |
| Conv47 | 256 | 2304 | 0.92 | 0.05 | 1.00 | 0.05 | 1.00 | 0.05 | 0.27 | 294.32 |
| Conv48 | 1024 | 256 | 0.86 | 0.05 | 1.00 | 0.05 | 1.00 | 0.05 | 13.24 | 1099.56 |
| Conv49 | 256 | 1024 | 0.82 | 0.05 | 1.00 | 0.05 | 1.00 | 0.05 | 0.38 | 294.32 |
| Conv50 | 256 | 2304 | 0.90 | 0.05 | 1.00 | 0.05 | 1.00 | 0.05 | 0.21 | 294.32 |
| Conv51 | 1024 | 256 | 0.90 | 0.05 | 1.00 | 0.05 | 1.00 | 0.05 | 4.57 | 1099.56 |
| Conv52 | 256 | 1024 | 0.88 | 0.05 | 1.00 | 0.05 | 1.00 | 0.05 | 0.2 | 294.32 |
| Conv53 | 256 | 2304 | 0.89 | 0.05 | 1.00 | 0.05 | 1.00 | 0.05 | 0.26 | 294.32 |
| Conv54 | 1024 | 256 | 0.88 | 0.05 | 1.00 | 0.05 | 1.00 | 0.05 | 8.23 | 1099.56 |
| Conv55 | 256 | 1024 | 0.89 | 0.05 | 1.00 | 0.05 | 1.00 | 0.05 | 0.28 | 294.32 |
| Conv56 | 256 | 2304 | 0.91 | 0.05 | 1.00 | 0.05 | 1.00 | 0.05 | 0.37 | 294.32 |
| Conv57 | 1024 | 256 | 0.85 | 0.05 | 1.00 | 0.05 | 1.00 | 0.05 | 8.72 | 1099.56 |
| Conv58 | 256 | 1024 | 0.88 | 0.05 | 1.00 | 0.05 | 1.00 | 0.05 | 0.28 | 294.32 |
| Conv59 | 256 | 2304 | 0.89 | 0.05 | 1.00 | 0.05 | 1.00 | 0.05 | 0.24 | 294.32 |
| Conv60 | 1024 | 256 | 0.89 | 0.05 | 1.00 | 0.05 | 1.00 | 0.05 | 5.31 | 1099.56 |
| Conv61 | 256 | 1024 | 0.89 | 0.05 | 1.00 | 0.05 | 1.00 | 0.05 | 0.25 | 294.32 |
| Conv62 | 256 | 2304 | 0.91 | 0.05 | 1.00 | 0.05 | 1.00 | 0.05 | 0.38 | 294.32 |
| Conv63 | 1024 | 256 | 0.89 | 0.05 | 1.00 | 0.05 | 1.00 | 0.05 | 5.22 | 1099.56 |
| Conv64 | 256 | 1024 | 0.90 | 0.05 | 1.00 | 0.05 | 1.00 | 0.05 | 0.21 | 294.32 |
| Conv65 | 256 | 2304 | 0.90 | 0.05 | 1.00 | 0.05 | 1.00 | 0.05 | 0.2 | 294.32 |
| Conv66 | 1024 | 256 | 0.90 | 0.05 | 1.00 | 0.05 | 1.00 | 0.05 | 2.98 | 1099.56 |
| Conv67 | 256 | 1024 | 0.92 | 0.05 | 1.00 | 0.05 | 1.00 | 0.05 | 0.26 | 294.32 |
| Conv68 | 256 | 2304 | 0.92 | 0.05 | 1.00 | 0.05 | 1.00 | 0.05 | 0.27 | 294.32 |
| Conv69 | 1024 | 256 | 0.90 | 0.05 | 1.00 | 0.05 | 1.00 | 0.05 | 4.05 | 1099.56 |
| Conv70 | 256 | 1024 | 0.87 | 0.05 | 1.00 | 0.05 | 1.00 | 0.05 | 0.28 | 294.32 |
| Conv71 | 256 | 2304 | 0.91 | 0.05 | 1.00 | 0.05 | 1.00 | 0.05 | 0.33 | 294.32 |
| Conv72 | 1024 | 256 | 0.91 | 0.05 | 1.00 | 0.05 | 1.00 | 0.05 | 8.2 | 1099.56 |
| Conv73 | 256 | 1024 | 0.85 | 0.05 | 1.00 | 0.05 | 1.00 | 0.05 | 0.31 | 294.32 |
| Conv74 | 256 | 2304 | 0.88 | 0.05 | 1.00 | 0.05 | 1.00 | 0.05 | 0.19 | 294.32 |
| Conv75 | 1024 | 256 | 0.88 | 0.05 | 1.00 | 0.05 | 1.00 | 0.05 | 5.23 | 1099.56 |
| Conv76 | 256 | 1024 | 0.86 | 0.05 | 1.00 | 0.05 | 1.00 | 0.05 | 0.3 | 294.32 |
| Conv77 | 256 | 2304 | 0.88 | 0.05 | 1.00 | 0.05 | 1.00 | 0.05 | 0.16 | 294.32 |
| Conv78 | 1024 | 256 | 0.88 | 0.05 | 1.00 | 0.05 | 1.00 | 0.05 | 2.86 | 1099.56 |
| Conv79 | 256 | 1024 | 0.90 | 0.05 | 1.00 | 0.05 | 1.00 | 0.05 | 0.3 | 294.32 |
| Conv80 | 256 | 2304 | 0.87 | 0.05 | 1.00 | 0.05 | 1.00 | 0.05 | 0.22 | 294.32 |
| Conv81 | 1024 | 256 | 0.88 | 0.05 | 1.00 | 0.05 | 1.00 | 0.05 | 2.53 | 1099.56 |
| Conv82 | 256 | 1024 | 0.89 | 0.05 | 1.00 | 0.05 | 1.00 | 0.05 | 0.2 | 294.32 |
| Conv83 | 256 | 2304 | 0.91 | 0.05 | 1.00 | 0.05 | 1.00 | 0.05 | 0.43 | 294.32 |
| Conv84 | 1024 | 256 | 0.90 | 0.05 | 1.00 | 0.05 | 1.00 | 0.05 | 5.84 | 1099.56 |
| Conv85 | 256 | 1024 | 0.90 | 0.05 | 1.00 | 0.05 | 1.00 | 0.05 | 0.28 | 294.32 |
| Conv86 | 256 | 2304 | 0.90 | 0.05 | 1.00 | 0.05 | 1.00 | 0.05 | 0.35 | 294.32 |
| Conv87 | 1024 | 256 | 0.85 | 0.05 | 1.00 | 0.05 | 1.00 | 0.05 | 3.08 | 1099.56 |
| Conv88 | 256 | 1024 | 0.87 | 0.05 | 1.00 | 0.05 | 1.00 | 0.05 | 0.25 | 294.32 |
| Conv89 | 256 | 2304 | 0.89 | 0.05 | 1.00 | 0.05 | 1.00 | 0.05 | 0.36 | 294.32 |
| Conv90 | 1024 | 256 | 0.91 | 0.05 | 1.00 | 0.05 | 1.00 | 0.05 | 3.51 | 1099.56 |
| Conv91 | 256 | 1024 | 0.91 | 0.05 | 1.00 | 0.05 | 1.00 | 0.05 | 0.26 | 294.32 |
| Conv92 | 256 | 2304 | 0.91 | 0.05 | 1.00 | 0.05 | 1.00 | 0.05 | 0.19 | 294.32 |
| Conv93 | 1024 | 256 | 0.90 | 0.05 | 1.00 | 0.05 | 1.00 | 0.05 | 3.62 | 1099.56 |
| Conv94 | 512 | 1024 | 0.82 | 0.05 | 1.00 | 0.05 | 1.00 | 0.05 | 0.89 | 565.75 |

| Layer | Number | dim | Gaussian p-value | Gaussian c-value | Mean_Left p-value | Mean_Left c-value | Mean_Right p-value | Mean_Right c-value | Sigma t-value | Sigma c-value |
|---|---|---|---|---|---|---|---|---|---|---|
| Conv95 | 512 | 4608 | 0.86 | 0.05 | 1.00 | 0.05 | 1.00 | 0.05 | 0.41 | 565.75 |
| Conv96 | 2048 | 512 | 0.82 | 0.05 | 1.00 | 0.05 | 1.00 | 0.05 | 4.89 | 2154.4 |
| Conv97 | 2048 | 1024 | 0.83 | 0.05 | 1.00 | 0.05 | 1.00 | 0.05 | 2.53 | 2154.4 |
| Conv98 | 512 | 2048 | 0.83 | 0.05 | 1.00 | 0.05 | 1.00 | 0.05 | 0.28 | 565.75 |
| Conv99 | 512 | 4608 | 0.91 | 0.05 | 1.00 | 0.05 | 1.00 | 0.05 | 0.26 | 565.75 |
| Conv100 | 2048 | 512 | 0.88 | 0.05 | 1.00 | 0.05 | 1.00 | 0.05 | 4.5 | 2154.4 |
| Conv101 | 512 | 2048 | 0.84 | 0.05 | 1.00 | 0.05 | 1.00 | 0.05 | 0.31 | 565.75 |
| Conv102 | 512 | 4608 | 0.90 | 0.05 | 1.00 | 0.05 | 1.00 | 0.05 | 0.19 | 565.75 |
| Conv103 | 2048 | 512 | 0.87 | 0.05 | 1.00 | 0.05 | 1.00 | 0.05 | 3.81 | 2154.4 |
| Conv104 | 512 | 9216 | 0.93 | 0.05 | 1.00 | 0.05 | 1.00 | 0.05 | 0.06 | 565.75 |
| Conv105 | 64 | 147 | 0.44 | 0.05 | 0.83 | 0.05 | 0.83 | 0.05 | 90.36 | 83.68 |
| **Passing rate** | - | - | 100.0% | | 100.0% | | 100.0% | | 99.05% | |

Table 42: PASCAL VOC 2007 ResNet101 bz=24

| Layer | Number | dim | Gaussian | | Mean_Left | | Mean_Right | | Sigma | |
|---|---|---|---|---|---|---|---|---|---|---|
| | | | p-value | c-value | p-value | c-value | p-value | c-value | t-value | c-value |
| Conv1 | 64 | 64 | 0.57 | 0.05 | 0.74 | 0.05 | 0.74 | 0.05 | 23.34 | 83.68 |
| Conv2 | 64 | 576 | 0.69 | 0.05 | 0.97 | 0.05 | 0.97 | 0.05 | 5.71 | 83.68 |
| Conv3 | 256 | 64 | 0.80 | 0.05 | 0.90 | 0.05 | 0.90 | 0.05 | 28.22 | 294.32 |
| Conv4 | 256 | 64 | 0.46 | 0.05 | 0.90 | 0.05 | 0.90 | 0.05 | 77.88 | 294.32 |
| Conv5 | 64 | 256 | 0.86 | 0.05 | 0.90 | 0.05 | 0.90 | 0.05 | 3.87 | 83.68 |
| Conv6 | 64 | 576 | 0.67 | 0.05 | 0.97 | 0.05 | 0.97 | 0.05 | 2.11 | 83.68 |
| Conv7 | 256 | 64 | 0.85 | 0.05 | 0.90 | 0.05 | 0.90 | 0.05 | 22.48 | 294.32 |
| Conv8 | 64 | 256 | 0.85 | 0.05 | 0.90 | 0.05 | 0.90 | 0.05 | 2.53 | 83.68 |
| Conv9 | 64 | 576 | 0.90 | 0.05 | 0.97 | 0.05 | 0.97 | 0.05 | 0.13 | 83.68 |
| Conv10 | 256 | 64 | 0.87 | 0.05 | 0.90 | 0.05 | 0.90 | 0.05 | 30.9 | 294.32 |
| Conv11 | 128 | 256 | 0.55 | 0.05 | 0.96 | 0.05 | 0.96 | 0.05 | 4.34 | 155.4 |
| Conv12 | 128 | 1152 | 0.68 | 0.05 | 1.00 | 0.05 | 1.00 | 0.05 | 2.69 | 155.4 |
| Conv13 | 512 | 128 | 0.70 | 0.05 | 0.99 | 0.05 | 0.99 | 0.05 | 28.4 | 565.75 |
| Conv14 | 512 | 256 | 0.56 | 0.05 | 1.00 | 0.05 | 1.00 | 0.05 | 14.13 | 565.75 |
| Conv15 | 128 | 512 | 0.89 | 0.05 | 0.99 | 0.05 | 0.99 | 0.05 | 0.6 | 155.4 |
| Conv16 | 128 | 1152 | 0.87 | 0.05 | 1.00 | 0.05 | 1.00 | 0.05 | 0.45 | 155.4 |
| Conv17 | 512 | 128 | 0.65 | 0.05 | 0.99 | 0.05 | 0.99 | 0.05 | 17.14 | 565.75 |
| Conv18 | 128 | 512 | 0.86 | 0.05 | 0.99 | 0.05 | 0.99 | 0.05 | 0.95 | 155.4 |
| Conv19 | 128 | 1152 | 0.89 | 0.05 | 1.00 | 0.05 | 1.00 | 0.05 | 0.39 | 155.4 |
| Conv20 | 512 | 128 | 0.81 | 0.05 | 0.99 | 0.05 | 0.99 | 0.05 | 16.68 | 565.75 |
| Conv21 | 128 | 512 | 0.85 | 0.05 | 0.99 | 0.05 | 0.99 | 0.05 | 1.02 | 155.4 |
| Conv22 | 128 | 1152 | 0.88 | 0.05 | 1.00 | 0.05 | 1.00 | 0.05 | 0.44 | 155.4 |
| Conv23 | 512 | 128 | 0.84 | 0.05 | 0.99 | 0.05 | 0.99 | 0.05 | 27.01 | 565.75 |
| Conv24 | 256 | 512 | 0.84 | 0.05 | 1.00 | 0.05 | 1.00 | 0.05 | 0.87 | 294.32 |
| Conv25 | 256 | 2304 | 0.88 | 0.05 | 1.00 | 0.05 | 1.00 | 0.05 | 0.5 | 294.32 |
| Conv26 | 1024 | 256 | 0.84 | 0.05 | 1.00 | 0.05 | 1.00 | 0.05 | 12.16 | 1099.56 |
| Conv27 | 1024 | 512 | 0.67 | 0.05 | 1.00 | 0.05 | 1.00 | 0.05 | 15.39 | 1099.56 |
| Conv28 | 256 | 1024 | 0.88 | 0.05 | 1.00 | 0.05 | 1.00 | 0.05 | 0.53 | 294.32 |
| Conv29 | 256 | 2304 | 0.90 | 0.05 | 1.00 | 0.05 | 1.00 | 0.05 | 0.7 | 294.32 |
| Conv30 | 1024 | 256 | 0.86 | 0.05 | 1.00 | 0.05 | 1.00 | 0.05 | 8.23 | 1099.56 |
| Conv31 | 256 | 1024 | 0.92 | 0.05 | 1.00 | 0.05 | 1.00 | 0.05 | 0.37 | 294.32 |
| Conv32 | 256 | 2304 | 0.93 | 0.05 | 1.00 | 0.05 | 1.00 | 0.05 | 0.31 | 294.32 |
| Conv33 | 1024 | 256 | 0.90 | 0.05 | 1.00 | 0.05 | 1.00 | 0.05 | 5.96 | 1099.56 |
| Conv34 | 256 | 1024 | 0.91 | 0.05 | 1.00 | 0.05 | 1.00 | 0.05 | 0.44 | 294.32 |
| Conv35 | 256 | 2304 | 0.92 | 0.05 | 1.00 | 0.05 | 1.00 | 0.05 | 0.19 | 294.32 |
| Conv36 | 1024 | 256 | 0.85 | 0.05 | 1.00 | 0.05 | 1.00 | 0.05 | 13.91 | 1099.56 |
| Conv37 | 256 | 1024 | 0.91 | 0.05 | 1.00 | 0.05 | 1.00 | 0.05 | 0.33 | 294.32 |
| Conv38 | 256 | 2304 | 0.93 | 0.05 | 1.00 | 0.05 | 1.00 | 0.05 | 0.22 | 294.32 |

| | | | | | | | | | | | |
|---|---|---|---|---|---|---|---|---|---|---|---|
| Conv39 | 1024 | 256 | 0.88 | 0.05 | 1.00 | 0.05 | 1.00 | 0.05 | 4.78 | 1099.56 |
| Conv40 | 256 | 1024 | 0.91 | 0.05 | 1.00 | 0.05 | 1.00 | 0.05 | 0.23 | 294.32 |
| Conv41 | 256 | 2304 | 0.93 | 0.05 | 1.00 | 0.05 | 1.00 | 0.05 | 0.19 | 294.32 |
| Conv42 | 1024 | 256 | 0.90 | 0.05 | 1.00 | 0.05 | 1.00 | 0.05 | 4.32 | 1099.56 |
| Conv43 | 256 | 1024 | 0.86 | 0.05 | 1.00 | 0.05 | 1.00 | 0.05 | 0.37 | 294.32 |
| Conv44 | 256 | 2304 | 0.91 | 0.05 | 1.00 | 0.05 | 1.00 | 0.05 | 0.27 | 294.32 |
| Conv45 | 1024 | 256 | 0.90 | 0.05 | 1.00 | 0.05 | 1.00 | 0.05 | 6.28 | 1099.56 |
| Conv46 | 256 | 1024 | 0.89 | 0.05 | 1.00 | 0.05 | 1.00 | 0.05 | 0.33 | 294.32 |
| Conv47 | 256 | 2304 | 0.91 | 0.05 | 1.00 | 0.05 | 1.00 | 0.05 | 0.27 | 294.32 |
| Conv48 | 1024 | 256 | 0.83 | 0.05 | 1.00 | 0.05 | 1.00 | 0.05 | 12.8 | 1099.56 |
| Conv49 | 256 | 1024 | 0.81 | 0.05 | 1.00 | 0.05 | 1.00 | 0.05 | 0.42 | 294.32 |
| Conv50 | 256 | 2304 | 0.91 | 0.05 | 1.00 | 0.05 | 1.00 | 0.05 | 0.17 | 294.32 |
| Conv51 | 1024 | 256 | 0.89 | 0.05 | 1.00 | 0.05 | 1.00 | 0.05 | 5.84 | 1099.56 |
| Conv52 | 256 | 1024 | 0.89 | 0.05 | 1.00 | 0.05 | 1.00 | 0.05 | 0.26 | 294.32 |
| Conv53 | 256 | 2304 | 0.83 | 0.05 | 1.00 | 0.05 | 1.00 | 0.05 | 0.22 | 294.32 |
| Conv54 | 1024 | 256 | 0.89 | 0.05 | 1.00 | 0.05 | 1.00 | 0.05 | 7.67 | 1099.56 |
| Conv55 | 256 | 1024 | 0.88 | 0.05 | 1.00 | 0.05 | 1.00 | 0.05 | 0.33 | 294.32 |
| Conv56 | 256 | 2304 | 0.91 | 0.05 | 1.00 | 0.05 | 1.00 | 0.05 | 0.3 | 294.32 |
| Conv57 | 1024 | 256 | 0.87 | 0.05 | 1.00 | 0.05 | 1.00 | 0.05 | 7.21 | 1099.56 |
| Conv58 | 256 | 1024 | 0.86 | 0.05 | 1.00 | 0.05 | 1.00 | 0.05 | 0.29 | 294.32 |
| Conv59 | 256 | 2304 | 0.91 | 0.05 | 1.00 | 0.05 | 1.00 | 0.05 | 0.16 | 294.32 |
| Conv60 | 1024 | 256 | 0.89 | 0.05 | 1.00 | 0.05 | 1.00 | 0.05 | 4.7 | 1099.56 |
| Conv61 | 256 | 1024 | 0.90 | 0.05 | 1.00 | 0.05 | 1.00 | 0.05 | 0.25 | 294.32 |
| Conv62 | 256 | 2304 | 0.92 | 0.05 | 1.00 | 0.05 | 1.00 | 0.05 | 0.26 | 294.32 |
| Conv63 | 1024 | 256 | 0.89 | 0.05 | 1.00 | 0.05 | 1.00 | 0.05 | 4.09 | 1099.56 |
| Conv64 | 256 | 1024 | 0.90 | 0.05 | 1.00 | 0.05 | 1.00 | 0.05 | 0.28 | 294.32 |
| Conv65 | 256 | 2304 | 0.92 | 0.05 | 1.00 | 0.05 | 1.00 | 0.05 | 0.16 | 294.32 |
| Conv66 | 1024 | 256 | 0.89 | 0.05 | 1.00 | 0.05 | 1.00 | 0.05 | 3.05 | 1099.56 |
| Conv67 | 256 | 1024 | 0.91 | 0.05 | 1.00 | 0.05 | 1.00 | 0.05 | 0.22 | 294.32 |
| Conv68 | 256 | 2304 | 0.91 | 0.05 | 1.00 | 0.05 | 1.00 | 0.05 | 0.16 | 294.32 |
| Conv69 | 1024 | 256 | 0.91 | 0.05 | 1.00 | 0.05 | 1.00 | 0.05 | 3.24 | 1099.56 |
| Conv70 | 256 | 1024 | 0.86 | 0.05 | 1.00 | 0.05 | 1.00 | 0.05 | 0.21 | 294.32 |
| Conv71 | 256 | 2304 | 0.93 | 0.05 | 1.00 | 0.05 | 1.00 | 0.05 | 0.17 | 294.32 |
| Conv72 | 1024 | 256 | 0.87 | 0.05 | 1.00 | 0.05 | 1.00 | 0.05 | 7.3 | 1099.56 |
| Conv73 | 256 | 1024 | 0.86 | 0.05 | 1.00 | 0.05 | 1.00 | 0.05 | 0.42 | 294.32 |
| Conv74 | 256 | 2304 | 0.88 | 0.05 | 1.00 | 0.05 | 1.00 | 0.05 | 0.11 | 294.32 |
| Conv75 | 1024 | 256 | 0.89 | 0.05 | 1.00 | 0.05 | 1.00 | 0.05 | 6.51 | 1099.56 |
| Conv76 | 256 | 1024 | 0.82 | 0.05 | 1.00 | 0.05 | 1.00 | 0.05 | 0.3 | 294.32 |
| Conv77 | 256 | 2304 | 0.89 | 0.05 | 1.00 | 0.05 | 1.00 | 0.05 | 0.09 | 294.32 |
| Conv78 | 1024 | 256 | 0.89 | 0.05 | 1.00 | 0.05 | 1.00 | 0.05 | 3.63 | 1099.56 |
| Conv79 | 256 | 1024 | 0.87 | 0.05 | 1.00 | 0.05 | 1.00 | 0.05 | 0.24 | 294.32 |
| Conv80 | 256 | 2304 | 0.89 | 0.05 | 1.00 | 0.05 | 1.00 | 0.05 | 0.09 | 294.32 |
| Conv81 | 1024 | 256 | 0.91 | 0.05 | 1.00 | 0.05 | 1.00 | 0.05 | 2.6 | 1099.56 |
| Conv82 | 256 | 1024 | 0.89 | 0.05 | 1.00 | 0.05 | 1.00 | 0.05 | 0.17 | 294.32 |
| Conv83 | 256 | 2304 | 0.92 | 0.05 | 1.00 | 0.05 | 1.00 | 0.05 | 0.19 | 294.32 |
| Conv84 | 1024 | 256 | 0.88 | 0.05 | 1.00 | 0.05 | 1.00 | 0.05 | 4.77 | 1099.56 |
| Conv85 | 256 | 1024 | 0.90 | 0.05 | 1.00 | 0.05 | 1.00 | 0.05 | 0.43 | 294.32 |
| Conv86 | 256 | 2304 | 0.92 | 0.05 | 1.00 | 0.05 | 1.00 | 0.05 | 0.14 | 294.32 |
| Conv87 | 1024 | 256 | 0.91 | 0.05 | 1.00 | 0.05 | 1.00 | 0.05 | 4.01 | 1099.56 |
| Conv88 | 256 | 1024 | 0.88 | 0.05 | 1.00 | 0.05 | 1.00 | 0.05 | 0.22 | 294.32 |
| Conv89 | 256 | 2304 | 0.90 | 0.05 | 1.00 | 0.05 | 1.00 | 0.05 | 0.14 | 294.32 |
| Conv90 | 1024 | 256 | 0.91 | 0.05 | 1.00 | 0.05 | 1.00 | 0.05 | 2.78 | 1099.56 |
| Conv91 | 256 | 1024 | 0.90 | 0.05 | 1.00 | 0.05 | 1.00 | 0.05 | 0.34 | 294.32 |
| Conv92 | 256 | 2304 | 0.87 | 0.05 | 1.00 | 0.05 | 1.00 | 0.05 | 0.1 | 294.32 |
| Conv93 | 1024 | 256 | 0.89 | 0.05 | 1.00 | 0.05 | 1.00 | 0.05 | 2.44 | 1099.56 |
| Conv94 | 512 | 1024 | 0.82 | 0.05 | 1.00 | 0.05 | 1.00 | 0.05 | 0.29 | 565.75 |
| Conv95 | 512 | 4608 | 0.84 | 0.05 | 1.00 | 0.05 | 1.00 | 0.05 | 0.11 | 565.75 |
| Conv96 | 2048 | 512 | 0.66 | 0.05 | 1.00 | 0.05 | 1.00 | 0.05 | 2.4 | 2154.4 |
| Conv97 | 2048 | 1024 | 0.53 | 0.05 | 1.00 | 0.05 | 1.00 | 0.05 | 2.25 | 2154.4 |

| Conv98 | 512 | 2048 | 0.81 | 0.05 | 1.00 | 0.05 | 1.00 | 0.05 | 0.12 | 565.75 |
| Conv99 | 512 | 4608 | 0.89 | 0.05 | 1.00 | 0.05 | 1.00 | 0.05 | 0.07 | 565.75 |
| Conv100 | 2048 | 512 | 0.85 | 0.05 | 1.00 | 0.05 | 1.00 | 0.05 | 2.43 | 2154.4 |
| Conv101 | 512 | 2048 | 0.64 | 0.05 | 1.00 | 0.05 | 1.00 | 0.05 | 0.2 | 565.75 |
| Conv102 | 512 | 4608 | 0.94 | 0.05 | 1.00 | 0.05 | 1.00 | 0.05 | 0.04 | 565.75 |
| Conv103 | 2048 | 512 | 0.90 | 0.05 | 1.00 | 0.05 | 1.00 | 0.05 | 2.01 | 2154.4 |
| Conv104 | 512 | 9216 | 0.95 | 0.05 | 1.00 | 0.05 | 1.00 | 0.05 | 0.0 | 565.75 |
| Conv105 | 64 | 147 | 0.44 | 0.05 | 0.83 | 0.05 | 0.83 | 0.05 | 90.36 | 83.68 |
| **Passing rate** | - | - | 100.0% | | 100.0% | | 100.0% | | 99.05% | |

Table 43: Visual Genome VGG16

| Layer | Number | dim | Gaussian | | Mean_Left | | Mean_Right | | Sigma | |
|---|---|---|---|---|---|---|---|---|---|---|
| | | | p-value | c-value | p-value | c-value | p-value | c-value | t-value | c-value |
| Conv1 | 64 | 27 | 0.50 | 0.05 | 0.66 | 0.05 | 0.66 | 0.05 | 147.34 | 83.68 |
| Conv2 | 64 | 576 | 0.85 | 0.05 | 0.97 | 0.05 | 0.97 | 0.05 | 3.2 | 83.68 |
| Conv3 | 128 | 576 | 0.68 | 0.05 | 1.00 | 0.05 | 1.00 | 0.05 | 6.85 | 155.4 |
| Conv4 | 128 | 1152 | 0.85 | 0.05 | 1.00 | 0.05 | 1.00 | 0.05 | 0.91 | 155.4 |
| Conv5 | 256 | 1152 | 0.71 | 0.05 | 1.00 | 0.05 | 1.00 | 0.05 | 0.92 | 294.32 |
| Conv6 | 256 | 2304 | 0.84 | 0.05 | 1.00 | 0.05 | 1.00 | 0.05 | 0.33 | 294.32 |
| Conv7 | 256 | 2304 | 0.81 | 0.05 | 1.00 | 0.05 | 1.00 | 0.05 | 0.27 | 294.32 |
| Conv8 | 512 | 2304 | 0.83 | 0.05 | 1.00 | 0.05 | 1.00 | 0.05 | 0.33 | 565.75 |
| Conv9 | 512 | 4608 | 0.88 | 0.05 | 1.00 | 0.05 | 1.00 | 0.05 | 0.11 | 565.75 |
| Conv10 | 512 | 4608 | 0.91 | 0.05 | 1.00 | 0.05 | 1.00 | 0.05 | 0.12 | 565.75 |
| Conv11 | 512 | 4608 | 0.87 | 0.05 | 1.00 | 0.05 | 1.00 | 0.05 | 0.1 | 565.75 |
| Conv12 | 512 | 4608 | 0.88 | 0.05 | 1.00 | 0.05 | 1.00 | 0.05 | 0.17 | 565.75 |
| Conv13 | 512 | 4608 | 0.88 | 0.05 | 1.00 | 0.05 | 1.00 | 0.05 | 0.6 | 565.75 |
| Conv14 | 512 | 4608 | 0.95 | 0.05 | 1.00 | 0.05 | 1.00 | 0.05 | 0.06 | 565.75 |
| Conv15 | 4096 | 25088 | 0.95 | 0.05 | 1.00 | 0.05 | 1.00 | 0.05 | 0.04 | 4246.0 |
| Conv16 | 4096 | 4096 | 0.95 | 0.05 | 1.00 | 0.05 | 1.00 | 0.05 | 0.15 | 4246.0 |
| **Passing rate** | - | - | 100.0% | | 100.0% | | 100.0% | | 93.75% | |

## P.7.3 IMAGE MATTING

config:

```
https://github.com/foamliu/Deep-Image-Matting-PyTorch
```

```
https://github.com/CDOTAD/AlphaGAN-Matting
```

Table 44: AlphaGAN matting

| Layer | Number | dim | Gaussian | | Mean_Left | | Mean_Right | | Sigma | |
|---|---|---|---|---|---|---|---|---|---|---|
| | | | p-value | c-value | p-value | c-value | p-value | c-value | t-value | c-value |
| Conv1 | 64 | 36 | 0.10 | 0.05 | 0.68 | 0.05 | 0.68 | 0.05 | 446.15 | 83.68 |
| Conv2 | 64 | 576 | 0.60 | 0.05 | 0.97 | 0.05 | 0.97 | 0.05 | 2.63 | 83.68 |
| Conv3 | 128 | 576 | 0.63 | 0.05 | 1.00 | 0.05 | 1.00 | 0.05 | 24.49 | 155.4 |
| Conv4 | 128 | 1152 | 0.70 | 0.05 | 1.00 | 0.05 | 1.00 | 0.05 | 2.71 | 155.4 |
| Conv5 | 256 | 1152 | 0.53 | 0.05 | 1.00 | 0.05 | 1.00 | 0.05 | 5.23 | 294.32 |
| Conv6 | 256 | 2304 | 0.85 | 0.05 | 1.00 | 0.05 | 1.00 | 0.05 | 0.81 | 294.32 |
| Conv7 | 256 | 2304 | 0.66 | 0.05 | 1.00 | 0.05 | 1.00 | 0.05 | 0.35 | 294.32 |
| Conv8 | 512 | 2304 | 0.67 | 0.05 | 1.00 | 0.05 | 1.00 | 0.05 | 0.78 | 565.75 |
| Conv9 | 512 | 4608 | 0.84 | 0.05 | 1.00 | 0.05 | 1.00 | 0.05 | 0.81 | 565.75 |

| Layer | Number | dim | | | | | | | | |
|-------|--------|------|------|------|------|------|------|------|------|------|
| Conv10 | 512 | 4608 | 0.87 | 0.05 | 1.00 | 0.05 | 1.00 | 0.05 | 0.87 | 565.75 |
| Conv11 | 512 | 4608 | 0.70 | 0.05 | 1.00 | 0.05 | 1.00 | 0.05 | 0.22 | 565.75 |
| Conv12 | 512 | 4608 | 0.81 | 0.05 | 1.00 | 0.05 | 1.00 | 0.05 | 0.18 | 565.75 |
| Conv13 | 512 | 4608 | 0.87 | 0.05 | 1.00 | 0.05 | 1.00 | 0.05 | 0.45 | 565.75 |
| Conv14 | 4096 | 25088 | 0.92 | 0.05 | 1.00 | 0.05 | 1.00 | 0.05 | 4.81 | 4246.0 |
| Conv15 | 512 | 4096 | 0.94 | 0.05 | 1.00 | 0.05 | 1.00 | 0.05 | 0.04 | 565.75 |
| Conv16 | 512 | 12800 | 0.93 | 0.05 | 1.00 | 0.05 | 1.00 | 0.05 | 0.09 | 565.75 |
| Conv17 | 256 | 12800 | 0.80 | 0.05 | 1.00 | 0.05 | 1.00 | 0.05 | 0.41 | 294.32 |
| Conv18 | 128 | 6400 | 0.70 | 0.05 | 1.00 | 0.05 | 1.00 | 0.05 | 0.87 | 155.4 |
| Conv19 | 64 | 3200 | 0.58 | 0.05 | 1.00 | 0.05 | 1.00 | 0.05 | 0.87 | 83.68 |
| Conv20 | 64 | 1600 | 0.46 | 0.05 | 1.00 | 0.05 | 1.00 | 0.05 | 2.43 | 83.68 |
| **Passing rate** | - | - | 100.0% | | 100.0% | | 100.0% | | 95.00% | |

Table 45: Deep image matting

| Layer | Number | dim | Gaussian | | Mean_Left | | Mean_Right | | Sigma | |
|-------|--------|------|----------|---------|-----------|---------|------------|---------|---------|---------|
| | | | p-value | c-value | p-value | c-value | p-value | c-value | t-value | c-value |
| Conv1 | 64 | 196 | 0.22 | 0.05 | 0.87 | 0.05 | 0.87 | 0.05 | 25.66 | 83.68 |
| Conv2 | 64 | 64 | 0.29 | 0.05 | 0.74 | 0.05 | 0.74 | 0.05 | 24.7 | 83.68 |
| Conv3 | 64 | 576 | 0.54 | 0.05 | 0.97 | 0.05 | 0.97 | 0.05 | 5.84 | 83.68 |
| Conv4 | 256 | 64 | 0.53 | 0.05 | 0.90 | 0.05 | 0.90 | 0.05 | 46.45 | 294.32 |
| Conv5 | 256 | 64 | 0.31 | 0.05 | 0.90 | 0.05 | 0.90 | 0.05 | 93.76 | 294.32 |
| Conv6 | 64 | 256 | 0.65 | 0.05 | 0.90 | 0.05 | 0.90 | 0.05 | 5.86 | 83.68 |
| Conv7 | 64 | 576 | 0.48 | 0.05 | 0.97 | 0.05 | 0.97 | 0.05 | 0.51 | 83.68 |
| Conv8 | 256 | 64 | 0.49 | 0.05 | 0.90 | 0.05 | 0.90 | 0.05 | 19.67 | 294.32 |
| Conv9 | 64 | 256 | 0.67 | 0.05 | 0.90 | 0.05 | 0.90 | 0.05 | 0.18 | 83.68 |
| Conv10 | 64 | 576 | 0.81 | 0.05 | 0.97 | 0.05 | 0.97 | 0.05 | 0.1 | 83.68 |
| Conv11 | 256 | 64 | 0.70 | 0.05 | 0.90 | 0.05 | 0.90 | 0.05 | 8.18 | 294.32 |
| Conv12 | 128 | 256 | 0.60 | 0.05 | 0.96 | 0.05 | 0.96 | 0.05 | 1.46 | 155.4 |
| Conv13 | 128 | 1152 | 0.67 | 0.05 | 1.00 | 0.05 | 1.00 | 0.05 | 0.16 | 155.4 |
| Conv14 | 512 | 128 | 0.60 | 0.05 | 0.99 | 0.05 | 0.99 | 0.05 | 89.07 | 565.75 |
| Conv15 | 512 | 256 | 0.27 | 0.05 | 1.00 | 0.05 | 1.00 | 0.05 | 83.22 | 565.75 |
| Conv16 | 128 | 512 | 0.39 | 0.05 | 0.99 | 0.05 | 0.99 | 0.05 | 2.41 | 155.4 |
| Conv17 | 128 | 1152 | 0.68 | 0.05 | 1.00 | 0.05 | 1.00 | 0.05 | 0.6 | 155.4 |
| Conv18 | 512 | 128 | 0.67 | 0.05 | 0.99 | 0.05 | 0.99 | 0.05 | 17.33 | 565.75 |
| Conv19 | 128 | 512 | 0.57 | 0.05 | 0.99 | 0.05 | 0.99 | 0.05 | 0.15 | 155.4 |
| Conv20 | 128 | 1152 | 0.61 | 0.05 | 1.00 | 0.05 | 1.00 | 0.05 | 0.09 | 155.4 |
| Conv21 | 512 | 128 | 0.52 | 0.05 | 0.99 | 0.05 | 0.99 | 0.05 | 3.61 | 565.75 |
| Conv22 | 128 | 512 | 0.64 | 0.05 | 0.99 | 0.05 | 0.99 | 0.05 | 0.12 | 155.4 |
| Conv23 | 128 | 1152 | 0.83 | 0.05 | 1.00 | 0.05 | 1.00 | 0.05 | 0.09 | 155.4 |
| Conv24 | 512 | 128 | 0.69 | 0.05 | 0.99 | 0.05 | 0.99 | 0.05 | 4.59 | 565.75 |
| Conv25 | 256 | 512 | 0.63 | 0.05 | 1.00 | 0.05 | 1.00 | 0.05 | 0.69 | 294.32 |
| Conv26 | 256 | 2304 | 0.67 | 0.05 | 1.00 | 0.05 | 1.00 | 0.05 | 0.1 | 294.32 |
| Conv27 | 1024 | 256 | 0.61 | 0.05 | 1.00 | 0.05 | 1.00 | 0.05 | 58.75 | 1099.56 |
| Conv28 | 1024 | 512 | 0.56 | 0.05 | 1.00 | 0.05 | 1.00 | 0.05 | 52.35 | 1099.56 |
| Conv29 | 256 | 1024 | 0.47 | 0.05 | 1.00 | 0.05 | 1.00 | 0.05 | 0.25 | 294.32 |
| Conv30 | 256 | 2304 | 0.55 | 0.05 | 1.00 | 0.05 | 1.00 | 0.05 | 0.39 | 294.32 |
| Conv31 | 1024 | 256 | 0.51 | 0.05 | 1.00 | 0.05 | 1.00 | 0.05 | 13.05 | 1099.56 |
| Conv32 | 256 | 1024 | 0.57 | 0.05 | 1.00 | 0.05 | 1.00 | 0.05 | 0.14 | 294.32 |
| Conv33 | 256 | 2304 | 0.80 | 0.05 | 1.00 | 0.05 | 1.00 | 0.05 | 0.14 | 294.32 |
| Conv34 | 1024 | 256 | 0.66 | 0.05 | 1.00 | 0.05 | 1.00 | 0.05 | 6.25 | 1099.56 |
| Conv35 | 256 | 1024 | 0.62 | 0.05 | 1.00 | 0.05 | 1.00 | 0.05 | 0.14 | 294.32 |
| Conv36 | 256 | 2304 | 0.64 | 0.05 | 1.00 | 0.05 | 1.00 | 0.05 | 0.19 | 294.32 |
| Conv37 | 1024 | 256 | 0.69 | 0.05 | 1.00 | 0.05 | 1.00 | 0.05 | 5.59 | 1099.56 |
| Conv38 | 256 | 1024 | 0.67 | 0.05 | 1.00 | 0.05 | 1.00 | 0.05 | 0.54 | 294.32 |

| Layer | | | | | | | | | | |
|---|---|---|---|---|---|---|---|---|---|---|
| Conv39 | 256 | 2304 | 0.69 | 0.05 | 1.00 | 0.05 | 1.00 | 0.05 | 0.68 | 294.32 |
| Conv40 | 1024 | 256 | 0.67 | 0.05 | 1.00 | 0.05 | 1.00 | 0.05 | 5.87 | 1099.56 |
| Conv41 | 256 | 1024 | 0.66 | 0.05 | 1.00 | 0.05 | 1.00 | 0.05 | 0.22 | 294.32 |
| Conv42 | 256 | 2304 | 0.80 | 0.05 | 1.00 | 0.05 | 1.00 | 0.05 | 0.21 | 294.32 |
| Conv43 | 1024 | 256 | 0.63 | 0.05 | 1.00 | 0.05 | 1.00 | 0.05 | 6.01 | 1099.56 |
| Conv44 | 512 | 1024 | 0.69 | 0.05 | 1.00 | 0.05 | 1.00 | 0.05 | 0.51 | 565.75 |
| Conv45 | 512 | 4608 | 0.64 | 0.05 | 1.00 | 0.05 | 1.00 | 0.05 | 0.28 | 565.75 |
| Conv46 | 2048 | 512 | 0.57 | 0.05 | 1.00 | 0.05 | 1.00 | 0.05 | 4.17 | 2154.4 |
| Conv47 | 2048 | 1024 | 0.38 | 0.05 | 1.00 | 0.05 | 1.00 | 0.05 | 2.83 | 2154.4 |
| Conv48 | 512 | 2048 | 0.28 | 0.05 | 1.00 | 0.05 | 1.00 | 0.05 | 3.58 | 565.75 |
| Conv49 | 512 | 4608 | 0.67 | 0.05 | 1.00 | 0.05 | 1.00 | 0.05 | 2.97 | 565.75 |
| Conv50 | 2048 | 512 | 0.70 | 0.05 | 1.00 | 0.05 | 1.00 | 0.05 | 17.54 | 2154.4 |
| Conv51 | 512 | 2048 | 0.55 | 0.05 | 1.00 | 0.05 | 1.00 | 0.05 | 6.14 | 565.75 |
| Conv52 | 512 | 4608 | 0.64 | 0.05 | 1.00 | 0.05 | 1.00 | 0.05 | 3.58 | 565.75 |
| Conv53 | 2048 | 512 | 0.61 | 0.05 | 1.00 | 0.05 | 1.00 | 0.05 | 24.98 | 2154.4 |
| Conv54 | 1024 | 2048 | 0.84 | 0.05 | 1.00 | 0.05 | 1.00 | 0.05 | 0.92 | 1099.56 |
| Conv55 | 1024 | 18432 | 0.66 | 0.05 | 1.00 | 0.05 | 1.00 | 0.05 | 3.88 | 1099.56 |
| Conv56 | 1024 | 18432 | 0.65 | 0.05 | 1.00 | 0.05 | 1.00 | 0.05 | 3.91 | 1099.56 |
| Conv57 | 1024 | 18432 | 0.68 | 0.05 | 1.00 | 0.05 | 1.00 | 0.05 | 2.13 | 1099.56 |
| Conv58 | 1024 | 5120 | 0.63 | 0.05 | 1.00 | 0.05 | 1.00 | 0.05 | 14.47 | 1099.56 |
| Conv59 | 1024 | 9216 | 0.24 | 0.05 | 1.00 | 0.05 | 1.00 | 0.05 | 48.34 | 1099.56 |
| Conv60 | 1024 | 1024 | 0.63 | 0.05 | 1.00 | 0.05 | 1.00 | 0.05 | 18.39 | 1099.56 |
| Conv61 | 1024 | 9216 | 0.55 | 0.05 | 1.00 | 0.05 | 1.00 | 0.05 | 59.85 | 1099.56 |
| Conv62 | 1024 | 11520 | 0.58 | 0.05 | 1.00 | 0.05 | 1.00 | 0.05 | 5.42 | 1099.56 |
| Conv63 | 512 | 9216 | 0.61 | 0.05 | 1.00 | 0.05 | 1.00 | 0.05 | 1.24 | 565.75 |
| Conv64 | 256 | 4608 | 0.62 | 0.05 | 1.00 | 0.05 | 1.00 | 0.05 | 0.73 | 294.32 |
| Conv65 | 64 | 2304 | 0.81 | 0.05 | 1.00 | 0.05 | 1.00 | 0.05 | 0.03 | 83.68 |
| Conv66 | 16 | 64 | 0.50 | 0.05 | 0.63 | 0.05 | 0.63 | 0.05 | 5.7 | 26.3 |
| Conv67 | 64 | 720 | 0.41 | 0.05 | 0.98 | 0.05 | 0.98 | 0.05 | 0.37 | 83.68 |
| Conv68 | 64 | 3136 | 0.37 | 0.05 | 1.00 | 0.05 | 1.00 | 0.05 | 0.9 | 83.68 |
| Conv69 | 64 | 576 | 0.39 | 0.05 | 0.97 | 0.05 | 0.97 | 0.05 | 6.03 | 83.68 |
| Conv70 | 64 | 603 | 0.32 | 0.05 | 0.98 | 0.05 | 0.98 | 0.05 | 17.08 | 83.68 |
| Conv71 | 64 | 576 | 0.23 | 0.05 | 0.97 | 0.05 | 0.97 | 0.05 | 46.48 | 83.68 |
| **Passing rate** | - | - | 100.0% | | 100.0% | | 100.0% | | 100.0% | |

## P.7.4 STYLE TRANSFER

config:

```
https://github.com/abhiskk/fast-neural-style
```

Table 46: Fast neural style (candy)

| Layer | Number | dim | Gaussian | | Mean_Left | | Mean_Right | | Sigma | |
|---|---|---|---|---|---|---|---|---|---|---|
| | | | p-value | c-value | p-value | c-value | p-value | c-value | t-value | c-value |
| Conv1 | 32 | 243 | 0.14 | 0.05 | 0.81 | 0.05 | 0.81 | 0.05 | 9.84 | 46.19 |
| Conv2 | 64 | 288 | 0.09 | 0.05 | 0.91 | 0.05 | 0.91 | 0.05 | 9.0 | 83.68 |
| Conv3 | 128 | 576 | 0.13 | 0.05 | 1.00 | 0.05 | 1.00 | 0.05 | 8.91 | 155.4 |
| Conv4 | 128 | 1152 | 0.02 | 0.05 | 1.00 | 0.05 | 1.00 | 0.05 | 23.03 | 155.4 |
| Conv5 | 128 | 1152 | 0.10 | 0.05 | 1.00 | 0.05 | 1.00 | 0.05 | 2.99 | 155.4 |
| Conv6 | 128 | 1152 | 0.11 | 0.05 | 1.00 | 0.05 | 1.00 | 0.05 | 1.76 | 155.4 |
| Conv7 | 128 | 1152 | 0.06 | 0.05 | 1.00 | 0.05 | 1.00 | 0.05 | 3.21 | 155.4 |
| Conv8 | 128 | 1152 | 0.13 | 0.05 | 1.00 | 0.05 | 1.00 | 0.05 | 19.79 | 155.4 |
| Conv9 | 128 | 1152 | 0.04 | 0.05 | 1.00 | 0.05 | 1.00 | 0.05 | 4.09 | 155.4 |
| Conv10 | 128 | 1152 | 0.24 | 0.05 | 1.00 | 0.05 | 1.00 | 0.05 | 1.11 | 155.4 |
| Conv11 | 128 | 1152 | 0.05 | 0.05 | 1.00 | 0.05 | 1.00 | 0.05 | 3.34 | 155.4 |

| Layer | Number | dim | | | | | | | | |
|---|---|---|---|---|---|---|---|---|---|---|
| Conv12 | 128 | 1152 | 0.15 | 0.05 | 1.00 | 0.05 | 1.00 | 0.05 | 1.49 | 155.4 |
| Conv13 | 128 | 1152 | 0.25 | 0.05 | 1.00 | 0.05 | 1.00 | 0.05 | 2.38 | 155.4 |
| Conv14 | 64 | 1152 | 0.30 | 0.05 | 1.00 | 0.05 | 1.00 | 0.05 | 0.93 | 83.68 |
| Conv15 | 32 | 576 | 0.31 | 0.05 | 0.91 | 0.05 | 0.91 | 0.05 | 1.08 | 46.19 |
| **Passing rate** | - | - | 86.67% | | 100.0% | | 100.0% | | 100.0% | |

Table 47: Fast neural style (mosaic)

| Layer | Number | dim | Gaussian | | Mean_Left | | Mean_Right | | Sigma | |
|---|---|---|---|---|---|---|---|---|---|---|
| | | | **p-value** | **c-value** | **p-value** | **c-value** | **p-value** | **c-value** | **t-value** | **c-value** |
| Conv1 | 32 | 243 | 0.18 | 0.05 | 0.81 | 0.05 | 0.81 | 0.05 | 1.44 | 46.19 |
| Conv2 | 64 | 288 | 0.23 | 0.05 | 0.91 | 0.05 | 0.91 | 0.05 | 7.95 | 83.68 |
| Conv3 | 128 | 576 | 0.21 | 0.05 | 1.00 | 0.05 | 1.00 | 0.05 | 27.77 | 155.4 |
| Conv4 | 128 | 1152 | 0.18 | 0.05 | 1.00 | 0.05 | 1.00 | 0.05 | 7.1 | 155.4 |
| Conv5 | 128 | 1152 | 0.04 | 0.05 | 1.00 | 0.05 | 1.00 | 0.05 | 2.78 | 155.4 |
| Conv6 | 128 | 1152 | 0.30 | 0.05 | 1.00 | 0.05 | 1.00 | 0.05 | 11.29 | 155.4 |
| Conv7 | 128 | 1152 | 0.13 | 0.05 | 1.00 | 0.05 | 1.00 | 0.05 | 3.61 | 155.4 |
| Conv8 | 128 | 1152 | 0.19 | 0.05 | 1.00 | 0.05 | 1.00 | 0.05 | 1.48 | 155.4 |
| Conv9 | 128 | 1152 | 0.12 | 0.05 | 1.00 | 0.05 | 1.00 | 0.05 | 2.61 | 155.4 |
| Conv10 | 128 | 1152 | 0.28 | 0.05 | 1.00 | 0.05 | 1.00 | 0.05 | 1.44 | 155.4 |
| Conv11 | 128 | 1152 | 0.23 | 0.05 | 1.00 | 0.05 | 1.00 | 0.05 | 2.27 | 155.4 |
| Conv12 | 128 | 1152 | 0.22 | 0.05 | 1.00 | 0.05 | 1.00 | 0.05 | 8.63 | 155.4 |
| Conv13 | 128 | 1152 | 0.26 | 0.05 | 1.00 | 0.05 | 1.00 | 0.05 | 2.33 | 155.4 |
| Conv14 | 64 | 1152 | 0.38 | 0.05 | 1.00 | 0.05 | 1.00 | 0.05 | 1.12 | 83.68 |
| Conv15 | 32 | 576 | 0.52 | 0.05 | 0.91 | 0.05 | 0.91 | 0.05 | 0.23 | 46.19 |
| **Passing rate** | - | - | 93.33% | | 100.0% | | 100.0% | | 100.0% | |

Table 48: Fast neural style (starry night)

| Layer | Number | dim | Gaussian | | Mean_Left | | Mean_Right | | Sigma | |
|---|---|---|---|---|---|---|---|---|---|---|
| | | | **p-value** | **c-value** | **p-value** | **c-value** | **p-value** | **c-value** | **t-value** | **c-value** |
| Conv1 | 32 | 243 | 0.13 | 0.05 | 0.81 | 0.05 | 0.81 | 0.05 | 9.92 | 46.19 |
| Conv2 | 64 | 288 | 0.14 | 0.05 | 0.91 | 0.05 | 0.91 | 0.05 | 19.39 | 83.68 |
| Conv3 | 128 | 576 | 0.20 | 0.05 | 1.00 | 0.05 | 1.00 | 0.05 | 32.68 | 155.4 |
| Conv4 | 128 | 1152 | 0.01 | 0.05 | 1.00 | 0.05 | 1.00 | 0.05 | 142.0 | 155.4 |
| Conv5 | 128 | 1152 | 0.02 | 0.05 | 1.00 | 0.05 | 1.00 | 0.05 | 28.99 | 155.4 |
| Conv6 | 128 | 1152 | 0.17 | 0.05 | 1.00 | 0.05 | 1.00 | 0.05 | 4.55 | 155.4 |
| Conv7 | 128 | 1152 | 0.10 | 0.05 | 1.00 | 0.05 | 1.00 | 0.05 | 3.05 | 155.4 |
| Conv8 | 128 | 1152 | 0.17 | 0.05 | 1.00 | 0.05 | 1.00 | 0.05 | 1.67 | 155.4 |
| Conv9 | 128 | 1152 | 0.07 | 0.05 | 1.00 | 0.05 | 1.00 | 0.05 | 2.42 | 155.4 |
| Conv10 | 128 | 1152 | 0.33 | 0.05 | 1.00 | 0.05 | 1.00 | 0.05 | 6.25 | 155.4 |
| Conv11 | 128 | 1152 | 0.12 | 0.05 | 1.00 | 0.05 | 1.00 | 0.05 | 2.06 | 155.4 |
| Conv12 | 128 | 1152 | 0.31 | 0.05 | 1.00 | 0.05 | 1.00 | 0.05 | 1.12 | 155.4 |
| Conv13 | 128 | 1152 | 0.16 | 0.05 | 1.00 | 0.05 | 1.00 | 0.05 | 1.58 | 155.4 |
| Conv14 | 64 | 1152 | 0.29 | 0.05 | 1.00 | 0.05 | 1.00 | 0.05 | 0.47 | 83.68 |
| Conv15 | 32 | 576 | 0.44 | 0.05 | 0.91 | 0.05 | 0.91 | 0.05 | 0.71 | 46.19 |
| **Passing rate** | - | - | 86.67% | | 100.0% | | 100.0% | | 100.0% | |

Table 49: Fast neural style (udnie)

| Layer | Number | dim | Gaussian | | Mean_Left | | Mean_Right | | Sigma | |
|---|---|---|---|---|---|---|---|---|---|---|
| | | | p-value | c-value | p-value | c-value | p-value | c-value | t-value | c-value |
| Conv1 | 32 | 243 | 0.22 | 0.05 | 0.81 | 0.05 | 0.81 | 0.05 | 9.61 | 46.19 |
| Conv2 | 64 | 288 | 0.17 | 0.05 | 0.91 | 0.05 | 0.91 | 0.05 | 13.05 | 83.68 |
| Conv3 | 128 | 576 | 0.19 | 0.05 | 1.00 | 0.05 | 1.00 | 0.05 | 10.46 | 155.4 |
| Conv4 | 128 | 1152 | 0.06 | 0.05 | 1.00 | 0.05 | 1.00 | 0.05 | 20.22 | 155.4 |
| Conv5 | 128 | 1152 | 0.14 | 0.05 | 1.00 | 0.05 | 1.00 | 0.05 | 4.46 | 155.4 |
| Conv6 | 128 | 1152 | 0.14 | 0.05 | 1.00 | 0.05 | 1.00 | 0.05 | 3.17 | 155.4 |
| Conv7 | 128 | 1152 | 0.02 | 0.05 | 1.00 | 0.05 | 1.00 | 0.05 | 5.85 | 155.4 |
| Conv8 | 128 | 1152 | 0.17 | 0.05 | 1.00 | 0.05 | 1.00 | 0.05 | 18.5 | 155.4 |
| Conv9 | 128 | 1152 | 0.00 | 0.05 | 1.00 | 0.05 | 1.00 | 0.05 | 18.88 | 155.4 |
| Conv10 | 128 | 1152 | 0.02 | 0.05 | 1.00 | 0.05 | 1.00 | 0.05 | 3.59 | 155.4 |
| Conv11 | 128 | 1152 | 0.00 | 0.05 | 1.00 | 0.05 | 1.00 | 0.05 | 10.35 | 155.4 |
| Conv12 | 128 | 1152 | 0.02 | 0.05 | 1.00 | 0.05 | 1.00 | 0.05 | 3.33 | 155.4 |
| Conv13 | 128 | 1152 | 0.21 | 0.05 | 1.00 | 0.05 | 1.00 | 0.05 | 3.99 | 155.4 |
| Conv14 | 64 | 1152 | 0.25 | 0.05 | 1.00 | 0.05 | 1.00 | 0.05 | 2.52 | 83.68 |
| Conv15 | 32 | 576 | 0.26 | 0.05 | 0.91 | 0.05 | 0.91 | 0.05 | 2.34 | 46.19 |
| **Passing rate** | - | - | 66.67% | | 100.0% | | 100.0% | | 100.0% | |

## P.7.5 GAN

config:

`https://github.com/csinva/gan-pretrained-pytorch`

Table 50: DCGAN MNIST

| Layer | Number | dim | Gaussian | | Mean_Left | | Mean_Right | | Sigma | |
|---|---|---|---|---|---|---|---|---|---|---|
| | | | p-value | c-value | p-value | c-value | p-value | c-value | t-value | c-value |
| Conv1 | 100 | 8192 | 0.70 | 0.05 | 1.00 | 0.05 | 1.00 | 0.05 | 1.88 | 124.34 |
| Conv2 | 512 | 4096 | 0.69 | 0.05 | 1.00 | 0.05 | 1.00 | 0.05 | 0.8 | 565.75 |
| Conv3 | 256 | 2048 | 0.64 | 0.05 | 1.00 | 0.05 | 1.00 | 0.05 | 0.92 | 294.32 |
| Conv4 | 128 | 1024 | 0.50 | 0.05 | 1.00 | 0.05 | 1.00 | 0.05 | 3.88 | 155.4 |
| **Passing rate** | - | - | 100.0% | | 100.0% | | 100.0% | | 100.0% | |

Table 51: DCGAN Cifar10

| Layer | Number | dim | Gaussian | | Mean_Left | | Mean_Right | | Sigma | |
|---|---|---|---|---|---|---|---|---|---|---|
| | | | p-value | c-value | p-value | c-value | p-value | c-value | t-value | c-value |
| Conv1 | 100 | 8192 | 0.65 | 0.05 | 1 | 0.05 | 1 | 0.05 | 1.48 | 124.34 |
| Conv2 | 512 | 4096 | 0.48 | 0.05 | 1 | 0.05 | 1 | 0.05 | 3.56 | 565.75 |
| Conv3 | 256 | 2048 | 0.32 | 0.05 | 1 | 0.05 | 1 | 0.05 | 10.52 | 294.32 |
| Conv4 | 128 | 1024 | 0.44 | 0.05 | 1 | 0.05 | 1 | 0.05 | 9.06 | 155.4 |
| **Passing rate** | - | - | 100.0% | | 100.0% | | 100.0% | | 100.0% | |

Table 52: DCGAN Cifar100

| Layer | Number | dim | Gaussian | | Mean_Left | | Mean_Right | | Sigma | |
|-------|--------|-----|----------|---------|-----------|---------|------------|---------|---------|---------|
| | | | p-value | c-value | p-value | c-value | p-value | c-value | t-value | c-value |
| Conv1 | 100 | 8192 | 0.53 | 0.05 | 1.00 | 0.05 | 1.00 | 0.05 | 2.13 | 124.34 |
| Conv2 | 512 | 4096 | 0.33 | 0.05 | 1.00 | 0.05 | 1.00 | 0.05 | 4.32 | 565.75 |
| Conv3 | 256 | 2048 | 0.48 | 0.05 | 1.00 | 0.05 | 1.00 | 0.05 | 8.49 | 294.32 |
| Conv4 | 128 | 1024 | 0.27 | 0.05 | 1.00 | 0.05 | 1.00 | 0.05 | 15.21 | 155.4 |
| **Passing rate** | - | - | 100.0% | | 100.0% | | 100.0% | | 100.0% | |

## P.8 BATCH NORMALIZATION

config:

`https://github.com/bearpaw/pytorch-classification.`

Table 53: Cifar10 VGG19

| Layer | Number | dim | Gaussian | | Mean_Left | | Mean_Right | | Sigma | |
|-------|--------|-----|----------|---------|-----------|---------|------------|---------|---------|---------|
| | | | p-value | c-value | p-value | c-value | p-value | c-value | t-value | c-value |
| Conv1 | 64 | 27 | 0.24 | 0.05 | 0.66 | 0.05 | 0.66 | 0.05 | 388.19 | 83.68 |
| Conv2 | 64 | 576 | 0.62 | 0.05 | 0.97 | 0.05 | 0.97 | 0.05 | 1.55 | 83.68 |
| Conv3 | 128 | 576 | 0.84 | 0.05 | 1.00 | 0.05 | 1.00 | 0.05 | 2.61 | 155.4 |
| Conv4 | 128 | 1152 | 0.86 | 0.05 | 1.00 | 0.05 | 1.00 | 0.05 | 0.39 | 155.4 |
| Conv5 | 256 | 1152 | 0.82 | 0.05 | 1.00 | 0.05 | 1.00 | 0.05 | 0.67 | 294.32 |
| Conv6 | 256 | 2304 | 0.88 | 0.05 | 1.00 | 0.05 | 1.00 | 0.05 | 0.41 | 294.32 |
| Conv7 | 256 | 2304 | 0.88 | 0.05 | 1.00 | 0.05 | 1.00 | 0.05 | 0.35 | 294.32 |
| Conv8 | 256 | 2304 | 0.90 | 0.05 | 1.00 | 0.05 | 1.00 | 0.05 | 0.21 | 294.32 |
| Conv9 | 512 | 2304 | 0.92 | 0.05 | 1.00 | 0.05 | 1.00 | 0.05 | 1.23 | 565.75 |
| Conv10 | 512 | 4608 | 0.91 | 0.05 | 1.00 | 0.05 | 1.00 | 0.05 | 1.08 | 565.75 |
| Conv11 | 512 | 4608 | 0.93 | 0.05 | 1.00 | 0.05 | 1.00 | 0.05 | 0.32 | 565.75 |
| Conv12 | 512 | 4608 | 0.94 | 0.05 | 1.00 | 0.05 | 1.00 | 0.05 | 0.16 | 565.75 |
| Conv13 | 512 | 4608 | 0.95 | 0.05 | 1.00 | 0.05 | 1.00 | 0.05 | 0.17 | 565.75 |
| Conv14 | 512 | 4608 | 0.95 | 0.05 | 1.00 | 0.05 | 1.00 | 0.05 | 0.17 | 565.75 |
| Conv15 | 512 | 4608 | 0.95 | 0.05 | 1.00 | 0.05 | 1.00 | 0.05 | 0.12 | 565.75 |
| Conv16 | 512 | 4608 | 0.93 | 0.05 | 1.00 | 0.05 | 1.00 | 0.05 | 0.25 | 565.75 |
| **Passing rate** | - | - | 100.0% | | 100.0% | | 100.0% | | 93.75% | |

Table 54: Cifar10 VGG19-bn

| Layer | Number | dim | Gaussian | | Mean_Left | | Mean_Right | | Sigma | |
|-------|--------|-----|----------|---------|-----------|---------|------------|---------|---------|---------|
| | | | p-value | c-value | p-value | c-value | p-value | c-value | t-value | c-value |
| Conv1 | 64 | 27 | 0.32 | 0.05 | 0.66 | 0.05 | 0.66 | 0.05 | 399.99 | 83.68 |
| Conv2 | 64 | 576 | 0.53 | 0.05 | 0.97 | 0.05 | 0.97 | 0.05 | 1.69 | 83.68 |
| Conv3 | 128 | 576 | 0.85 | 0.05 | 1.00 | 0.05 | 1.00 | 0.05 | 3.16 | 155.4 |
| Conv4 | 128 | 1152 | 0.86 | 0.05 | 1.00 | 0.05 | 1.00 | 0.05 | 0.45 | 155.4 |
| Conv5 | 256 | 1152 | 0.84 | 0.05 | 1.00 | 0.05 | 1.00 | 0.05 | 1.05 | 294.32 |
| Conv6 | 256 | 2304 | 0.88 | 0.05 | 1.00 | 0.05 | 1.00 | 0.05 | 0.38 | 294.32 |
| Conv7 | 256 | 2304 | 0.90 | 0.05 | 1.00 | 0.05 | 1.00 | 0.05 | 0.45 | 294.32 |
| Conv8 | 256 | 2304 | 0.91 | 0.05 | 1.00 | 0.05 | 1.00 | 0.05 | 0.28 | 294.32 |
| Conv9 | 512 | 2304 | 0.91 | 0.05 | 1.00 | 0.05 | 1.00 | 0.05 | 1.6 | 565.75 |
| Conv10 | 512 | 4608 | 0.94 | 0.05 | 1.00 | 0.05 | 1.00 | 0.05 | 0.91 | 565.75 |
| Conv11 | 512 | 4608 | 0.92 | 0.05 | 1.00 | 0.05 | 1.00 | 0.05 | 0.22 | 565.75 |
| Conv12 | 512 | 4608 | 0.94 | 0.05 | 1.00 | 0.05 | 1.00 | 0.05 | 0.18 | 565.75 |

| Conv13 | 512 | 4608 | 0.94 | 0.05 | 1.00 | 0.05 | 1.00 | 0.05 | 0.29 | 565.75 |
| Conv14 | 512 | 4608 | 0.94 | 0.05 | 1.00 | 0.05 | 1.00 | 0.05 | 0.23 | 565.75 |
| Conv15 | 512 | 4608 | 0.95 | 0.05 | 1.00 | 0.05 | 1.00 | 0.05 | 0.13 | 565.75 |
| Conv16 | 512 | 4608 | 0.93 | 0.05 | 1.00 | 0.05 | 1.00 | 0.05 | 0.32 | 565.75 |
| **Passing rate** | - | - | 100.0% | | 100.0% | | 100.0% | | 93.75% | |

## P.9 PYTORCH PRETRAIN

config: `http://pytorch.org/docs/master/torchvision/index.html`.

Table 55: Pytorch pre-trained VGG11

| Layer | Number | dim | Gaussian | | Mean_Left | | Mean_Right | | Sigma | |
|---|---|---|---|---|---|---|---|---|---|---|
| | | | p-value | c-value | p-value | c-value | p-value | c-value | t-value | c-value |
| Conv1 | 64 | 27 | 0.08 | 0.05 | 0.98 | 0.05 | 0.98 | 0.05 | 18556.94 | 83.68 |
| Conv2 | 128 | 576 | 0.43 | 0.05 | 1.00 | 0.05 | 1.00 | 0.05 | 365.45 | 155.4 |
| Conv3 | 256 | 1152 | 0.55 | 0.05 | 1.00 | 0.05 | 1.00 | 0.05 | 118.19 | 294.32 |
| Conv4 | 256 | 2304 | 0.42 | 0.05 | 1.00 | 0.05 | 1.00 | 0.05 | 60.64 | 294.32 |
| Conv5 | 512 | 2304 | 0.67 | 0.05 | 1.00 | 0.05 | 1.00 | 0.05 | 44.8 | 565.75 |
| Conv6 | 512 | 4608 | 0.83 | 0.05 | 1.00 | 0.05 | 1.00 | 0.05 | 23.01 | 565.75 |
| Conv7 | 512 | 4608 | 0.82 | 0.05 | 1.00 | 0.05 | 1.00 | 0.05 | 16.13 | 565.75 |
| Conv8 | 512 | 4608 | 0.88 | 0.05 | 1.00 | 0.05 | 1.00 | 0.05 | 43.86 | 565.75 |
| **Passing rate** | - | - | 100.0% | | 100.0% | | 100.0% | | 75.0% | |

Table 56: Pytorch pre-trained VGG16

| Layer | Number | dim | Gaussian | | Mean_Left | | Mean_Right | | Sigma | |
|---|---|---|---|---|---|---|---|---|---|---|
| | | | p-value | c-value | p-value | c-value | p-value | c-value | t-value | c-value |
| Conv1 | 64 | 27 | 0.20 | 0.05 | 0.98 | 0.05 | 0.98 | 0.05 | 8106.7 | 83.68 |
| Conv2 | 64 | 576 | 0.59 | 0.05 | 1.00 | 0.05 | 1.00 | 0.05 | 134.8 | 83.68 |
| Conv3 | 128 | 576 | 0.55 | 0.05 | 1.00 | 0.05 | 1.00 | 0.05 | 152.47 | 155.4 |
| Conv4 | 128 | 1152 | 0.81 | 0.05 | 1.00 | 0.05 | 1.00 | 0.05 | 64.62 | 155.4 |
| Conv5 | 256 | 1152 | 0.56 | 0.05 | 1.00 | 0.05 | 1.00 | 0.05 | 40.31 | 294.32 |
| Conv6 | 256 | 2304 | 0.70 | 0.05 | 1.00 | 0.05 | 1.00 | 0.05 | 21.44 | 294.32 |
| Conv7 | 256 | 2304 | 0.55 | 0.05 | 1.00 | 0.05 | 1.00 | 0.05 | 25.05 | 294.32 |
| Conv8 | 512 | 2304 | 0.64 | 0.05 | 1.00 | 0.05 | 1.00 | 0.05 | 33.34 | 565.75 |
| Conv9 | 512 | 4608 | 0.85 | 0.05 | 1.00 | 0.05 | 1.00 | 0.05 | 11.92 | 565.75 |
| Conv10 | 512 | 4608 | 0.88 | 0.05 | 1.00 | 0.05 | 1.00 | 0.05 | 11.49 | 565.75 |
| Conv11 | 512 | 4608 | 0.83 | 0.05 | 1.00 | 0.05 | 1.00 | 0.05 | 9.21 | 565.75 |
| Conv12 | 512 | 4608 | 0.87 | 0.05 | 1.00 | 0.05 | 1.00 | 0.05 | 8.74 | 565.75 |
| Conv13 | 512 | 4608 | 0.92 | 0.05 | 1.00 | 0.05 | 1.00 | 0.05 | 33.55 | 565.75 |
| **Passing rate** | - | - | 100.0% | | 100.0% | | 100.0% | | 84.62% | |

Table 57: Pytorch pre-trained VGG19

| Layer | Number | dim | Gaussian | | Mean_Left | | Mean_Right | | Sigma | |
|---|---|---|---|---|---|---|---|---|---|---|

| | | | p-value | c-value | p-value | c-value | p-value | c-value | t-value | c-value |
|---|---|---|---|---|---|---|---|---|---|---|
| Conv1 | 64 | 27 | 0.25 | 0.05 | 0.98 | 0.05 | 0.98 | 0.05 | 6679.05 | 83.68 |
| Conv2 | 64 | 576 | 0.61 | 0.05 | 1.00 | 0.05 | 1.00 | 0.05 | 163.5 | 83.68 |
| Conv3 | 128 | 576 | 0.58 | 0.05 | 1.00 | 0.05 | 1.00 | 0.05 | 132.27 | 155.4 |
| Conv4 | 128 | 1152 | 0.58 | 0.05 | 1.00 | 0.05 | 1.00 | 0.05 | 52.71 | 155.4 |
| Conv5 | 256 | 1152 | 0.51 | 0.05 | 1.00 | 0.05 | 1.00 | 0.05 | 36.91 | 294.32 |
| Conv6 | 256 | 2304 | 0.66 | 0.05 | 1.00 | 0.05 | 1.00 | 0.05 | 19.93 | 294.32 |
| Conv7 | 256 | 2304 | 0.85 | 0.05 | 1.00 | 0.05 | 1.00 | 0.05 | 14.54 | 294.32 |
| Conv8 | 256 | 2304 | 0.64 | 0.05 | 1.00 | 0.05 | 1.00 | 0.05 | 20.24 | 294.32 |
| Conv9 | 512 | 2304 | 0.64 | 0.05 | 1.00 | 0.05 | 1.00 | 0.05 | 26.91 | 565.75 |
| Conv10 | 512 | 4608 | 0.86 | 0.05 | 1.00 | 0.05 | 1.00 | 0.05 | 10.7 | 565.75 |
| Conv11 | 512 | 4608 | 0.89 | 0.05 | 1.00 | 0.05 | 1.00 | 0.05 | 8.98 | 565.75 |
| Conv12 | 512 | 4608 | 0.89 | 0.05 | 1.00 | 0.05 | 1.00 | 0.05 | 7.87 | 565.75 |
| Conv13 | 512 | 4608 | 0.87 | 0.05 | 1.00 | 0.05 | 1.00 | 0.05 | 7.51 | 565.75 |
| Conv14 | 512 | 4608 | 0.90 | 0.05 | 1.00 | 0.05 | 1.00 | 0.05 | 9.71 | 565.75 |
| Conv15 | 512 | 4608 | 0.92 | 0.05 | 1.00 | 0.05 | 1.00 | 0.05 | 7.61 | 565.75 |
| Conv16 | 512 | 4608 | 0.90 | 0.05 | 1.00 | 0.05 | 1.00 | 0.05 | 22.02 | 565.75 |
| **Passing rate** | - | - | 100.0% | | 100.0% | | 100.0% | | 87.50% | |

Table 58: Pytorch pre-trained ResNet18

| Layer | Number | dim | Gaussian | | Mean_Left | | Mean_Right | | Sigma | |
|---|---|---|---|---|---|---|---|---|---|---|
| | | | p-value | c-value | p-value | c-value | p-value | c-value | t-value | c-value |
| Conv1 | 64 | 147 | 0.31 | 0.05 | 1.00 | 0.05 | 1.00 | 0.05 | 3000.89 | 83.68 |
| Conv2 | 64 | 576 | 0.42 | 0.05 | 1.00 | 0.05 | 1.00 | 0.05 | 90.98 | 83.68 |
| Conv3 | 64 | 576 | 0.63 | 0.05 | 1.00 | 0.05 | 1.00 | 0.05 | 42.7 | 83.68 |
| Conv4 | 64 | 576 | 0.52 | 0.05 | 1.00 | 0.05 | 1.00 | 0.05 | 82.51 | 83.68 |
| Conv5 | 64 | 576 | 0.70 | 0.05 | 1.00 | 0.05 | 1.00 | 0.05 | 32.27 | 83.68 |
| Conv6 | 128 | 576 | 0.82 | 0.05 | 1.00 | 0.05 | 1.00 | 0.05 | 14.37 | 155.4 |
| Conv7 | 128 | 1152 | 0.67 | 0.05 | 1.00 | 0.05 | 1.00 | 0.05 | 35.48 | 155.4 |
| Conv8 | 128 | 1152 | 0.66 | 0.05 | 1.00 | 0.05 | 1.00 | 0.05 | 15.09 | 155.4 |
| Conv9 | 128 | 1152 | 0.81 | 0.05 | 1.00 | 0.05 | 1.00 | 0.05 | 34.26 | 155.4 |
| Conv10 | 256 | 1152 | 0.70 | 0.05 | 1.00 | 0.05 | 1.00 | 0.05 | 11.25 | 294.32 |
| Conv11 | 256 | 2304 | 0.82 | 0.05 | 1.00 | 0.05 | 1.00 | 0.05 | 10.09 | 294.32 |
| Conv12 | 256 | 2304 | 0.84 | 0.05 | 1.00 | 0.05 | 1.00 | 0.05 | 12.4 | 294.32 |
| Conv13 | 256 | 2304 | 0.82 | 0.05 | 1.00 | 0.05 | 1.00 | 0.05 | 42.29 | 294.32 |
| Conv14 | 512 | 2304 | 0.70 | 0.05 | 1.00 | 0.05 | 1.00 | 0.05 | 4.8 | 565.75 |
| Conv15 | 512 | 4608 | 0.81 | 0.05 | 1.00 | 0.05 | 1.00 | 0.05 | 8.93 | 565.75 |
| Conv16 | 512 | 4608 | 0.86 | 0.05 | 1.00 | 0.05 | 1.00 | 0.05 | 3.92 | 565.75 |
| Conv17 | 512 | 4608 | 0.85 | 0.05 | 1.00 | 0.05 | 1.00 | 0.05 | 1.47 | 565.75 |
| **Passing rate** | - | - | 100.0% | | 100.0% | | 100.0% | | 88.24% | |

Table 59: Pytorch pre-trained ResNet34

| Layer | Number | dim | Gaussian | | Mean_Left | | Mean_Right | | Sigma | |
|---|---|---|---|---|---|---|---|---|---|---|
| | | | p-value | c-value | p-value | c-value | p-value | c-value | t-value | c-value |
| Conv1 | 64 | 147 | 0.35 | 0.05 | 1.00 | 0.05 | 1.00 | 0.05 | 3020.16 | 83.68 |
| Conv2 | 64 | 576 | 0.54 | 0.05 | 1.00 | 0.05 | 1.00 | 0.05 | 76.24 | 83.68 |
| Conv3 | 64 | 576 | 0.67 | 0.05 | 1.00 | 0.05 | 1.00 | 0.05 | 40.95 | 83.68 |

| Layer | Number | dim | Gaussian p-value | Gaussian c-value | Mean_Left p-value | Mean_Left c-value | Mean_Right p-value | Mean_Right c-value | Sigma t-value | Sigma c-value |
|---|---|---|---|---|---|---|---|---|---|---|
| Conv4 | 64 | 576 | 0.67 | 0.05 | 1.00 | 0.05 | 1.00 | 0.05 | 26.53 | 83.68 |
| Conv5 | 64 | 576 | 0.84 | 0.05 | 1.00 | 0.05 | 1.00 | 0.05 | 33.62 | 83.68 |
| Conv6 | 64 | 576 | 0.84 | 0.05 | 1.00 | 0.05 | 1.00 | 0.05 | 14.46 | 83.68 |
| Conv7 | 64 | 576 | 0.83 | 0.05 | 1.00 | 0.05 | 1.00 | 0.05 | 41.19 | 83.68 |
| Conv8 | 128 | 576 | 0.83 | 0.05 | 1.00 | 0.05 | 1.00 | 0.05 | 13.86 | 155.4 |
| Conv9 | 128 | 1152 | 0.81 | 0.05 | 1.00 | 0.05 | 1.00 | 0.05 | 86.4 | 155.4 |
| Conv10 | 128 | 1152 | 0.66 | 0.05 | 1.00 | 0.05 | 1.00 | 0.05 | 20.55 | 155.4 |
| Conv11 | 128 | 1152 | 0.85 | 0.05 | 1.00 | 0.05 | 1.00 | 0.05 | 36.72 | 155.4 |
| Conv12 | 128 | 1152 | 0.54 | 0.05 | 1.00 | 0.05 | 1.00 | 0.05 | 9.64 | 155.4 |
| Conv13 | 128 | 1152 | 0.84 | 0.05 | 1.00 | 0.05 | 1.00 | 0.05 | 18.48 | 155.4 |
| Conv14 | 128 | 1152 | 0.69 | 0.05 | 1.00 | 0.05 | 1.00 | 0.05 | 11.52 | 155.4 |
| Conv15 | 128 | 1152 | 0.85 | 0.05 | 1.00 | 0.05 | 1.00 | 0.05 | 39.51 | 155.4 |
| Conv16 | 256 | 1152 | 0.83 | 0.05 | 1.00 | 0.05 | 1.00 | 0.05 | 9.06 | 294.32 |
| Conv17 | 256 | 2304 | 0.81 | 0.05 | 1.00 | 0.05 | 1.00 | 0.05 | 13.99 | 294.32 |
| Conv18 | 256 | 2304 | 0.81 | 0.05 | 1.00 | 0.05 | 1.00 | 0.05 | 15.44 | 294.32 |
| Conv19 | 256 | 2304 | 0.64 | 0.05 | 1.00 | 0.05 | 1.00 | 0.05 | 17.56 | 294.32 |
| Conv20 | 256 | 2304 | 0.68 | 0.05 | 1.00 | 0.05 | 1.00 | 0.05 | 7.88 | 294.32 |
| Conv21 | 256 | 2304 | 0.69 | 0.05 | 1.00 | 0.05 | 1.00 | 0.05 | 29.32 | 294.32 |
| Conv22 | 256 | 2304 | 0.87 | 0.05 | 1.00 | 0.05 | 1.00 | 0.05 | 7.87 | 294.32 |
| Conv23 | 256 | 2304 | 0.87 | 0.05 | 1.00 | 0.05 | 1.00 | 0.05 | 45.76 | 294.32 |
| Conv24 | 256 | 2304 | 0.86 | 0.05 | 1.00 | 0.05 | 1.00 | 0.05 | 6.93 | 294.32 |
| Conv25 | 256 | 2304 | 0.84 | 0.05 | 1.00 | 0.05 | 1.00 | 0.05 | 36.02 | 294.32 |
| Conv26 | 256 | 2304 | 0.89 | 0.05 | 1.00 | 0.05 | 1.00 | 0.05 | 6.78 | 294.32 |
| Conv27 | 256 | 2304 | 0.89 | 0.05 | 1.00 | 0.05 | 1.00 | 0.05 | 22.54 | 294.32 |
| Conv28 | 512 | 2304 | 0.83 | 0.05 | 1.00 | 0.05 | 1.00 | 0.05 | 2.31 | 565.75 |
| Conv29 | 512 | 4608 | 0.86 | 0.05 | 1.00 | 0.05 | 1.00 | 0.05 | 2.42 | 565.75 |
| Conv30 | 512 | 4608 | 0.85 | 0.05 | 1.00 | 0.05 | 1.00 | 0.05 | 4.27 | 565.75 |
| Conv31 | 512 | 4608 | 0.86 | 0.05 | 1.00 | 0.05 | 1.00 | 0.05 | 6.3 | 565.75 |
| Conv32 | 512 | 4608 | 0.88 | 0.05 | 1.00 | 0.05 | 1.00 | 0.05 | 1.64 | 565.75 |
| Conv33 | 512 | 4608 | 0.87 | 0.05 | 1.00 | 0.05 | 1.00 | 0.05 | 1.24 | 565.75 |
| **Passing rate** | - | - | 100.0% | | 100.0% | | 100.0% | | 96.97% | |

Table 60: Pytorch pre-trained ResNet50

| Layer | Number | dim | Gaussian p-value | Gaussian c-value | Mean_Left p-value | Mean_Left c-value | Mean_Right p-value | Mean_Right c-value | Sigma t-value | Sigma c-value |
|---|---|---|---|---|---|---|---|---|---|---|
| Conv1 | **64** | **147** | 0.43 | 0.05 | 1.00 | 0.05 | 1.00 | 0.05 | 1512.0 | 83.68 |
| Conv2 | **64** | **64** | 0.47 | 0.05 | 1.00 | 0.05 | 1.00 | 0.05 | 512.72 | 83.68 |
| Conv3 | **64** | **576** | 0.64 | 0.05 | 1.00 | 0.05 | 1.00 | 0.05 | 103.97 | 83.68 |
| Conv4 | **256** | **64** | 0.69 | 0.05 | 1.00 | 0.05 | 1.00 | 0.05 | 786.24 | 294.32 |
| Conv5 | 64 | 256 | 0.86 | 0.05 | 1.00 | 0.05 | 1.00 | 0.05 | 52.01 | 83.68 |
| Conv6 | 64 | 576 | 0.60 | 0.05 | 1.00 | 0.05 | 1.00 | 0.05 | 31.52 | 83.68 |
| Conv7 | **256** | **64** | 0.84 | 0.05 | 1.00 | 0.05 | 1.00 | 0.05 | 619.68 | 294.32 |
| Conv8 | 64 | 256 | 0.89 | 0.05 | 1.00 | 0.05 | 1.00 | 0.05 | 14.19 | 83.68 |
| Conv9 | 64 | 576 | 0.85 | 0.05 | 1.00 | 0.05 | 1.00 | 0.05 | 6.07 | 83.68 |
| Conv10 | **256** | **64** | 0.85 | 0.05 | 1.00 | 0.05 | 1.00 | 0.05 | 789.85 | 294.32 |
| Conv11 | 128 | 256 | 0.81 | 0.05 | 1.00 | 0.05 | 1.00 | 0.05 | 49.33 | 155.4 |
| Conv12 | 128 | 1152 | 0.84 | 0.05 | 1.00 | 0.05 | 1.00 | 0.05 | 6.6 | 155.4 |
| Conv13 | **512** | **128** | 0.70 | 0.05 | 1.00 | 0.05 | 1.00 | 0.05 | 1154.78 | 565.75 |
| Conv14 | 128 | 512 | 0.86 | 0.05 | 1.00 | 0.05 | 1.00 | 0.05 | 20.26 | 155.4 |
| Conv15 | 128 | 1152 | 0.84 | 0.05 | 1.00 | 0.05 | 1.00 | 0.05 | 59.31 | 155.4 |
| Conv16 | **512** | **128** | 0.84 | 0.05 | 1.00 | 0.05 | 1.00 | 0.05 | 897.22 | 565.75 |
| Conv17 | 128 | 512 | 0.87 | 0.05 | 1.00 | 0.05 | 1.00 | 0.05 | 14.87 | 155.4 |
| Conv18 | 128 | 1152 | 0.86 | 0.05 | 1.00 | 0.05 | 1.00 | 0.05 | 10.56 | 155.4 |

| Layer | Number | dim | | | | | | | | |
|-------|--------|-----|------|------|------|------|------|------|--------|---------|
| Conv19 | 512 | 128 | 0.81 | 0.05 | 1.00 | 0.05 | 1.00 | 0.05 | 326.18 | 565.75 |
| Conv20 | 128 | 512 | 0.85 | 0.05 | 1.00 | 0.05 | 1.00 | 0.05 | 5.57 | 155.4 |
| Conv21 | 128 | 1152 | 0.88 | 0.05 | 1.00 | 0.05 | 1.00 | 0.05 | 7.15 | 155.4 |
| Conv22 | 512 | 128 | 0.84 | 0.05 | 1.00 | 0.05 | 1.00 | 0.05 | 511.13 | 565.75 |
| Conv23 | 256 | 512 | 0.82 | 0.05 | 1.00 | 0.05 | 1.00 | 0.05 | 32.44 | 294.32 |
| Conv24 | 256 | 2304 | 0.89 | 0.05 | 1.00 | 0.05 | 1.00 | 0.05 | 8.35 | 294.32 |
| Conv25 | 1024 | 256 | 0.83 | 0.05 | 1.00 | 0.05 | 1.00 | 0.05 | 608.91 | 1099.56 |
| Conv26 | 256 | 1024 | 0.84 | 0.05 | 1.00 | 0.05 | 1.00 | 0.05 | 10.33 | 294.32 |
| Conv27 | 256 | 2304 | 0.86 | 0.05 | 1.00 | 0.05 | 1.00 | 0.05 | 12.19 | 294.32 |
| Conv28 | 1024 | 256 | 0.62 | 0.05 | 1.00 | 0.05 | 1.00 | 0.05 | 365.91 | 1099.56 |
| Conv29 | 256 | 1024 | 0.85 | 0.05 | 1.00 | 0.05 | 1.00 | 0.05 | 6.6 | 294.32 |
| Conv30 | 256 | 2304 | 0.89 | 0.05 | 1.00 | 0.05 | 1.00 | 0.05 | 7.69 | 294.32 |
| Conv31 | 1024 | 256 | 0.81 | 0.05 | 1.00 | 0.05 | 1.00 | 0.05 | 306.61 | 1099.56 |
| Conv32 | 256 | 1024 | 0.87 | 0.05 | 1.00 | 0.05 | 1.00 | 0.05 | 5.97 | 294.32 |
| Conv33 | 256 | 2304 | 0.85 | 0.05 | 1.00 | 0.05 | 1.00 | 0.05 | 6.47 | 294.32 |
| Conv34 | 1024 | 256 | 0.83 | 0.05 | 1.00 | 0.05 | 1.00 | 0.05 | 312.36 | 1099.56 |
| Conv35 | 256 | 1024 | 0.85 | 0.05 | 1.00 | 0.05 | 1.00 | 0.05 | 5.81 | 294.32 |
| Conv36 | 256 | 2304 | 0.89 | 0.05 | 1.00 | 0.05 | 1.00 | 0.05 | 5.38 | 294.32 |
| Conv37 | 1024 | 256 | 0.83 | 0.05 | 1.00 | 0.05 | 1.00 | 0.05 | 330.13 | 1099.56 |
| Conv38 | 256 | 1024 | 0.69 | 0.05 | 1.00 | 0.05 | 1.00 | 0.05 | 4.9 | 294.32 |
| Conv39 | 256 | 2304 | 0.88 | 0.05 | 1.00 | 0.05 | 1.00 | 0.05 | 4.16 | 294.32 |
| Conv40 | 1024 | 256 | 0.82 | 0.05 | 1.00 | 0.05 | 1.00 | 0.05 | 319.79 | 1099.56 |
| Conv41 | 512 | 1024 | 0.82 | 0.05 | 1.00 | 0.05 | 1.00 | 0.05 | 6.42 | 565.75 |
| Conv42 | 512 | 4608 | 0.69 | 0.05 | 1.00 | 0.05 | 1.00 | 0.05 | 1.38 | 565.75 |
| Conv43 | 2048 | 512 | 0.81 | 0.05 | 1.00 | 0.05 | 1.00 | 0.05 | 97.84 | 2154.4 |
| Conv44 | 512 | 2048 | 0.48 | 0.05 | 1.00 | 0.05 | 1.00 | 0.05 | 3.4 | 565.75 |
| Conv45 | 512 | 4608 | 0.88 | 0.05 | 1.00 | 0.05 | 1.00 | 0.05 | 2.7 | 565.75 |
| Conv46 | 2048 | 512 | 0.87 | 0.05 | 1.00 | 0.05 | 1.00 | 0.05 | 106.86 | 2154.4 |
| Conv47 | 512 | 2048 | 0.65 | 0.05 | 1.00 | 0.05 | 1.00 | 0.05 | 4.34 | 565.75 |
| Conv48 | 512 | 4608 | 0.92 | 0.05 | 1.00 | 0.05 | 1.00 | 0.05 | 0.65 | 565.75 |
| Conv49 | 2048 | 512 | 0.85 | 0.05 | 1.00 | 0.05 | 1.00 | 0.05 | 139.29 | 2154.4 |
| **Passing rate** | - | - | 100.0% | | 100.0% | | 100.0% | | 83.67% | |

## P.10 LEARNING RATE

config:

`https://github.com/bearpaw/pytorch-classification.`

Table 61: Cifar10 VGG19 bn schedule(82-164)

| Layer | Number | dim | Gaussian | | Mean_Left | | Mean_Right | | Sigma | |
|-------|--------|-----|---------|---------|-----------|---------|------------|---------|---------|---------|
| | | | p-value | c-value | p-value | c-value | p-value | c-value | t-value | c-value |
| Conv1 | 64 | 27 | 0.22 | 0.05 | 0.66 | 0.05 | 0.66 | 0.05 | 313.52 | 83.68 |
| Conv2 | 64 | 576 | 0.62 | 0.05 | 0.97 | 0.05 | 0.97 | 0.05 | 1.52 | 83.68 |
| Conv3 | 128 | 576 | 0.82 | 0.05 | 1.00 | 0.05 | 1.00 | 0.05 | 1.88 | 155.4 |
| Conv4 | 128 | 1152 | 0.89 | 0.05 | 1.00 | 0.05 | 1.00 | 0.05 | 0.31 | 155.4 |
| Conv5 | 256 | 1152 | 0.88 | 0.05 | 1.00 | 0.05 | 1.00 | 0.05 | 0.5 | 294.32 |
| Conv6 | 256 | 2304 | 0.91 | 0.05 | 1.00 | 0.05 | 1.00 | 0.05 | 0.24 | 294.32 |
| Conv7 | 256 | 2304 | 0.90 | 0.05 | 1.00 | 0.05 | 1.00 | 0.05 | 0.22 | 294.32 |
| Conv8 | 256 | 2304 | 0.90 | 0.05 | 1.00 | 0.05 | 1.00 | 0.05 | 0.13 | 294.32 |
| Conv9 | 512 | 2304 | 0.91 | 0.05 | 1.00 | 0.05 | 1.00 | 0.05 | 0.6 | 565.75 |
| Conv10 | 512 | 4608 | 0.92 | 0.05 | 1.00 | 0.05 | 1.00 | 0.05 | 0.4 | 565.75 |
| Conv11 | 512 | 4608 | 0.89 | 0.05 | 1.00 | 0.05 | 1.00 | 0.05 | 0.53 | 565.75 |
| Conv12 | 512 | 4608 | 0.95 | 0.05 | 1.00 | 0.05 | 1.00 | 0.05 | 0.32 | 565.75 |

| Layer | Number | dim | | | | | | | | |
|-------|--------|-----|------|------|------|------|------|------|------|--------|
| Conv13 | 512 | 4608 | 0.95 | 0.05 | 1.00 | 0.05 | 1.00 | 0.05 | 0.19 | 565.75 |
| Conv14 | 512 | 4608 | 0.95 | 0.05 | 1.00 | 0.05 | 1.00 | 0.05 | 0.11 | 565.75 |
| Conv15 | 512 | 4608 | 0.95 | 0.05 | 1.00 | 0.05 | 1.00 | 0.05 | 0.08 | 565.75 |
| Conv16 | 512 | 4608 | 0.94 | 0.05 | 1.00 | 0.05 | 1.00 | 0.05 | 0.32 | 565.75 |
| **Passing rate** | - | - | 100.0% | | 100.0% | | 100.0% | | 93.75% | |

Table 62: Cifar10 VGG19 bn schedule(60-120)

| Layer | Number | dim | Gaussian | | Mean_Left | | Mean_Right | | Sigma | |
|-------|--------|-----|----------|----------|-----------|----------|------------|----------|---------|---------|
| | | | p-value | c-value | p-value | c-value | p-value | c-value | t-value | c-value |
| Conv1 | 64 | 27 | 0.23 | 0.05 | 0.66 | 0.05 | 0.66 | 0.05 | 292.85 | 83.68 |
| Conv2 | 64 | 576 | 0.64 | 0.05 | 0.97 | 0.05 | 0.97 | 0.05 | 1.18 | 83.68 |
| Conv3 | 128 | 576 | 0.85 | 0.05 | 1.00 | 0.05 | 1.00 | 0.05 | 2.03 | 155.4 |
| Conv4 | 128 | 1152 | 0.88 | 0.05 | 1.00 | 0.05 | 1.00 | 0.05 | 0.34 | 155.4 |
| Conv5 | 256 | 1152 | 0.85 | 0.05 | 1.00 | 0.05 | 1.00 | 0.05 | 0.54 | 294.32 |
| Conv6 | 256 | 2304 | 0.90 | 0.05 | 1.00 | 0.05 | 1.00 | 0.05 | 0.3 | 294.32 |
| Conv7 | 256 | 2304 | 0.91 | 0.05 | 1.00 | 0.05 | 1.00 | 0.05 | 0.23 | 294.32 |
| Conv8 | 256 | 2304 | 0.91 | 0.05 | 1.00 | 0.05 | 1.00 | 0.05 | 0.2 | 294.32 |
| Conv9 | 512 | 2304 | 0.91 | 0.05 | 1.00 | 0.05 | 1.00 | 0.05 | 0.65 | 565.75 |
| Conv10 | 512 | 4608 | 0.92 | 0.05 | 1.00 | 0.05 | 1.00 | 0.05 | 0.38 | 565.75 |
| Conv11 | 512 | 4608 | 0.92 | 0.05 | 1.00 | 0.05 | 1.00 | 0.05 | 0.57 | 565.75 |
| Conv12 | 512 | 4608 | 0.94 | 0.05 | 1.00 | 0.05 | 1.00 | 0.05 | 0.34 | 565.75 |
| Conv13 | 512 | 4608 | 0.95 | 0.05 | 1.00 | 0.05 | 1.00 | 0.05 | 0.17 | 565.75 |
| Conv14 | 512 | 4608 | 0.95 | 0.05 | 1.00 | 0.05 | 1.00 | 0.05 | 0.11 | 565.75 |
| Conv15 | 512 | 4608 | 0.95 | 0.05 | 1.00 | 0.05 | 1.00 | 0.05 | 0.08 | 565.75 |
| Conv16 | 512 | 4608 | 0.94 | 0.05 | 1.00 | 0.05 | 1.00 | 0.05 | 0.26 | 565.75 |
| **Passing rate** | - | - | 100.0% | | 100.0% | | 100.0% | | 93.75% | |

Table 63: Cifar10 VGG19 bn coslr

| Layer | Number | dim | Gaussian | | Mean_Left | | Mean_Right | | Sigma | |
|-------|--------|-----|----------|----------|-----------|----------|------------|----------|---------|---------|
| | | | p-value | c-value | p-value | c-value | p-value | c-value | t-value | c-value |
| Conv1 | 64 | 27 | 0.35 | 0.05 | 0.66 | 0.05 | 0.66 | 0.05 | 313.71 | 83.68 |
| Conv2 | 64 | 576 | 0.65 | 0.05 | 0.97 | 0.05 | 0.97 | 0.05 | 1.31 | 83.68 |
| Conv3 | 128 | 576 | 0.81 | 0.05 | 1.00 | 0.05 | 1.00 | 0.05 | 2.38 | 155.4 |
| Conv4 | 128 | 1152 | 0.89 | 0.05 | 1.00 | 0.05 | 1.00 | 0.05 | 0.34 | 155.4 |
| Conv5 | 256 | 1152 | 0.87 | 0.05 | 1.00 | 0.05 | 1.00 | 0.05 | 0.44 | 294.32 |
| Conv6 | 256 | 2304 | 0.90 | 0.05 | 1.00 | 0.05 | 1.00 | 0.05 | 0.3 | 294.32 |
| Conv7 | 256 | 2304 | 0.90 | 0.05 | 1.00 | 0.05 | 1.00 | 0.05 | 0.22 | 294.32 |
| Conv8 | 256 | 2304 | 0.85 | 0.05 | 1.00 | 0.05 | 1.00 | 0.05 | 0.1 | 294.32 |
| Conv9 | 512 | 2304 | 0.89 | 0.05 | 1.00 | 0.05 | 1.00 | 0.05 | 0.74 | 565.75 |
| Conv10 | 512 | 4608 | 0.91 | 0.05 | 1.00 | 0.05 | 1.00 | 0.05 | 0.34 | 565.75 |
| Conv11 | 512 | 4608 | 0.92 | 0.05 | 1.00 | 0.05 | 1.00 | 0.05 | 0.54 | 565.75 |
| Conv12 | 512 | 4608 | 0.95 | 0.05 | 1.00 | 0.05 | 1.00 | 0.05 | 0.32 | 565.75 |
| Conv13 | 512 | 4608 | 0.95 | 0.05 | 1.00 | 0.05 | 1.00 | 0.05 | 0.2 | 565.75 |
| Conv14 | 512 | 4608 | 0.95 | 0.05 | 1.00 | 0.05 | 1.00 | 0.05 | 0.11 | 565.75 |
| Conv15 | 512 | 4608 | 0.95 | 0.05 | 1.00 | 0.05 | 1.00 | 0.05 | 0.08 | 565.75 |
| Conv16 | 512 | 4608 | 0.93 | 0.05 | 1.00 | 0.05 | 1.00 | 0.05 | 0.32 | 565.75 |
| **Passing rate** | - | - | 100.0% | | 100.0% | | 100.0% | | 93.75% | |

# Q ADDITIONAL EXPERIMENTS FOR SECTION 2

In this section, we use more networks (Pytorch Pretrain Models[9] to verify the obersevations in Fig. 2 (a-d). We show the results of three layers for each network.

## Q.1 VGG16

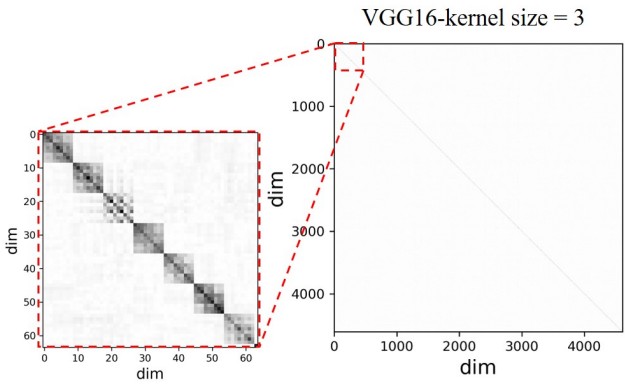

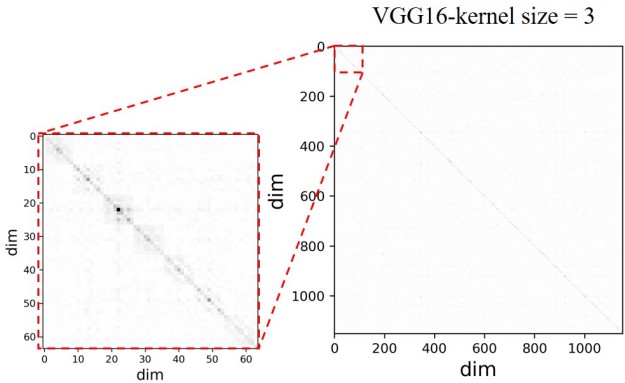

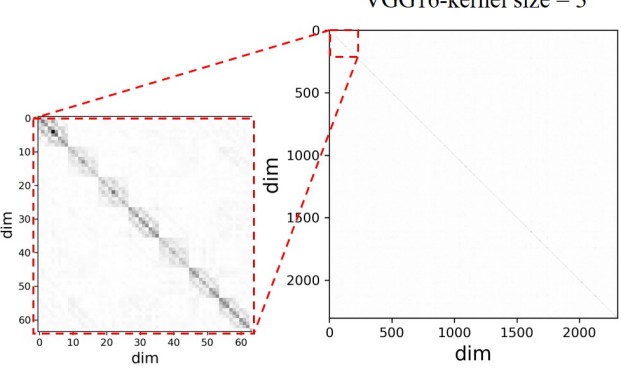

[9]https://pytorch.org/docs/stable/torchvision/models.html

## Q.2    VGG19

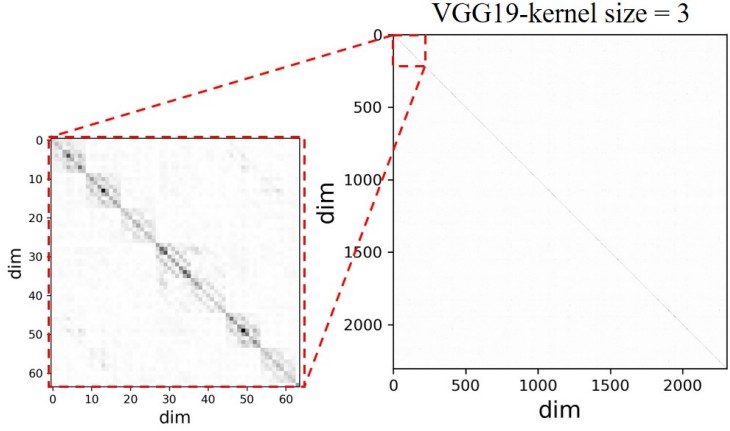

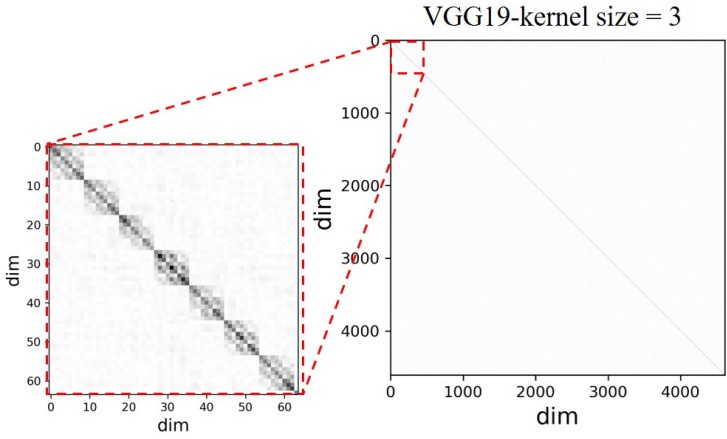

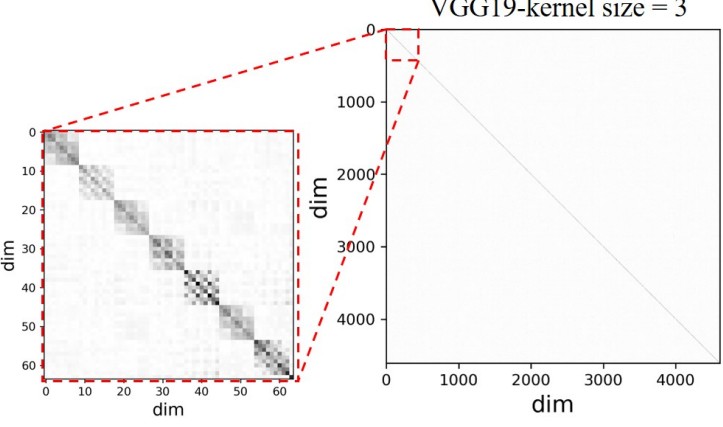

## Q.3 RESNET18

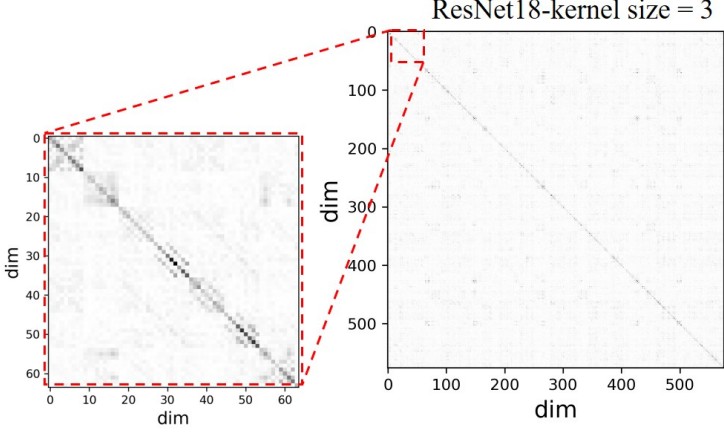

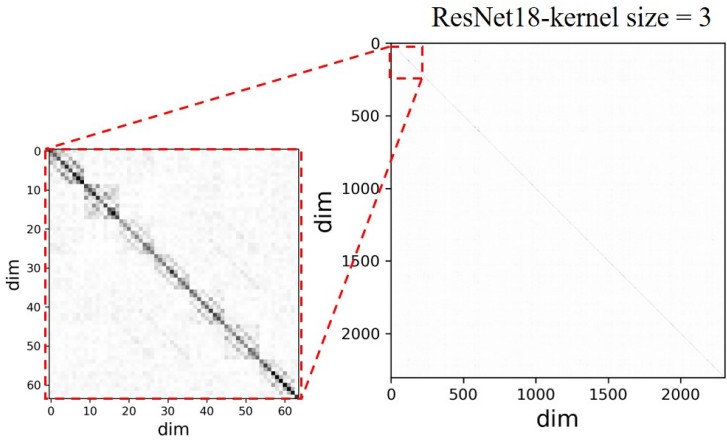

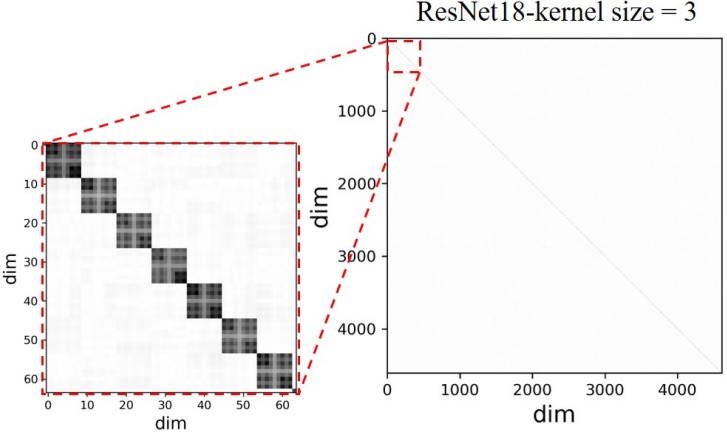

## Q.4 RESNET50

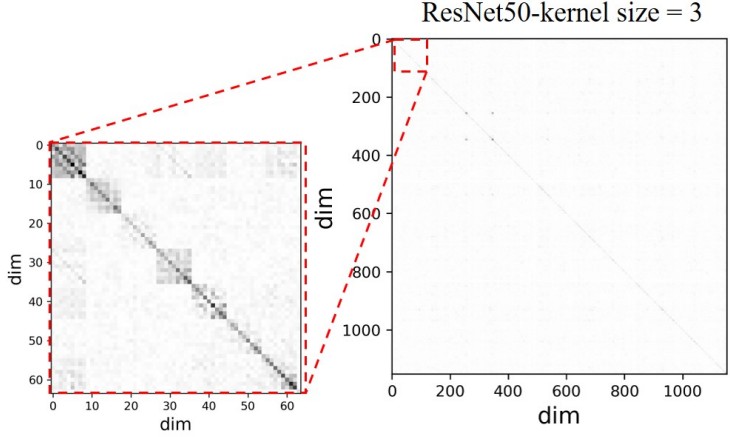

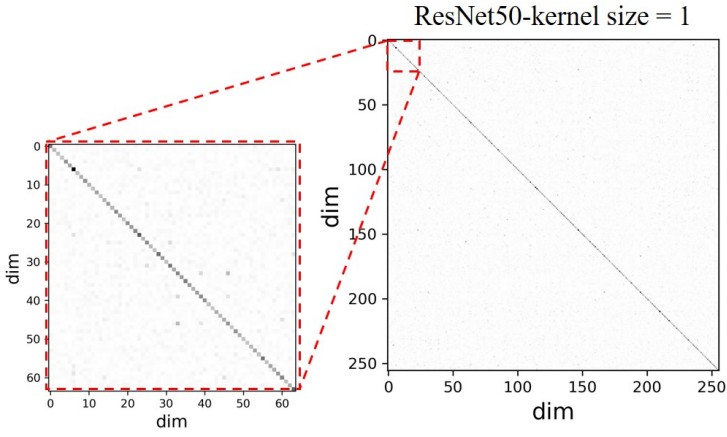

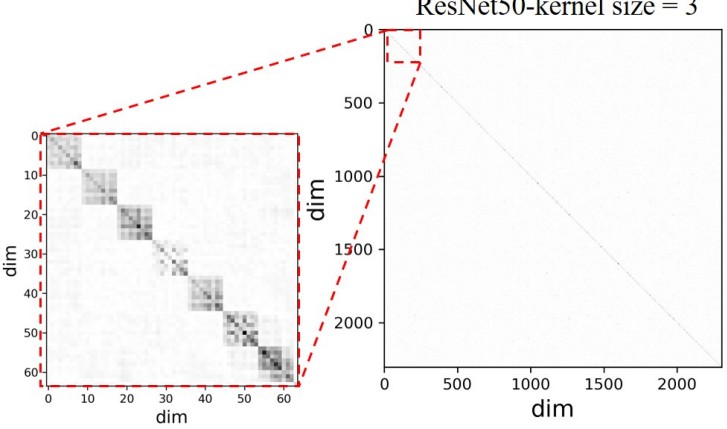

## Q.5   ALEXNET

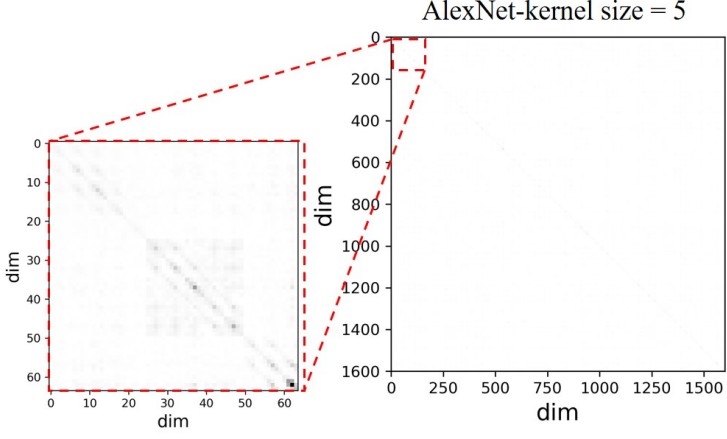

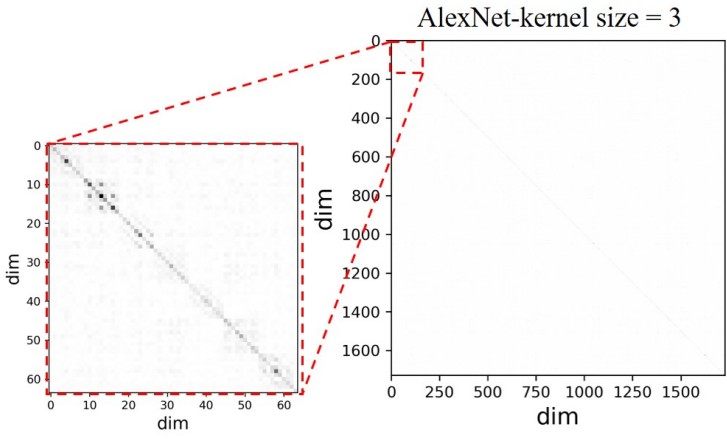

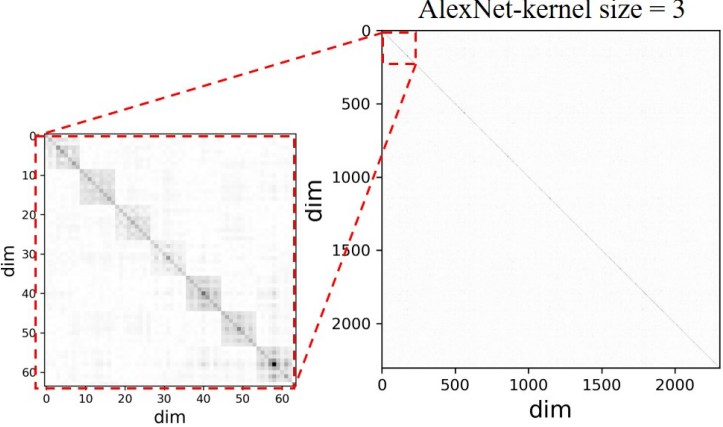

## Q.6   DENSENET

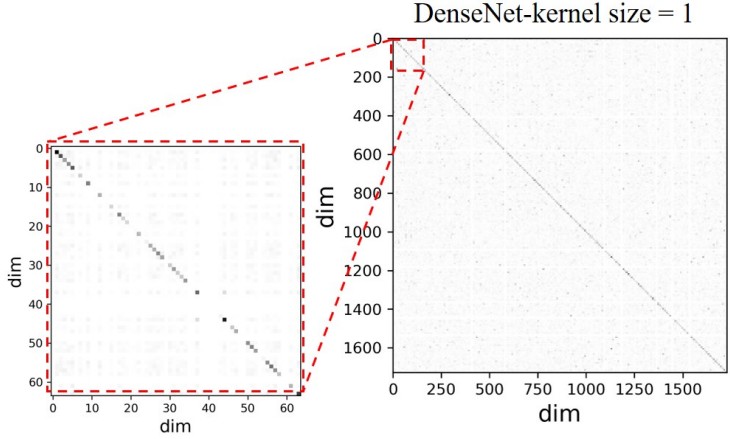

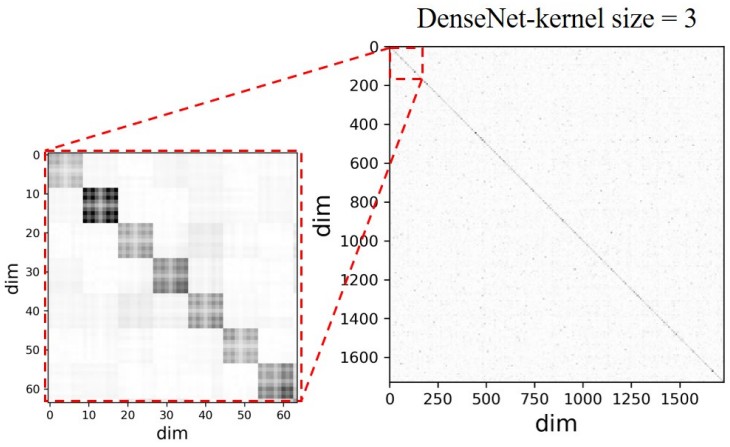

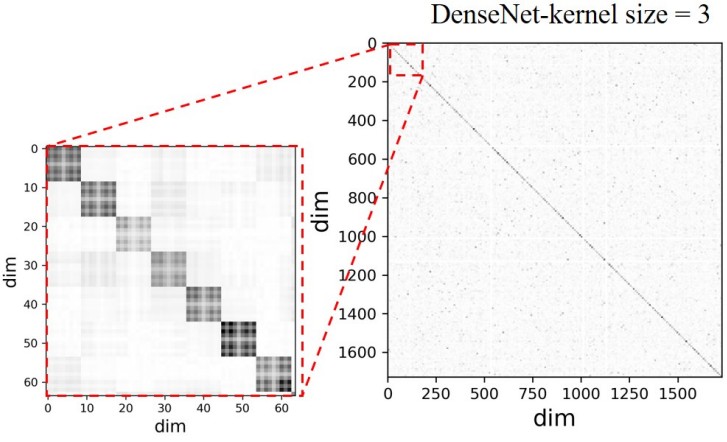

## Q.7    RESNEXT

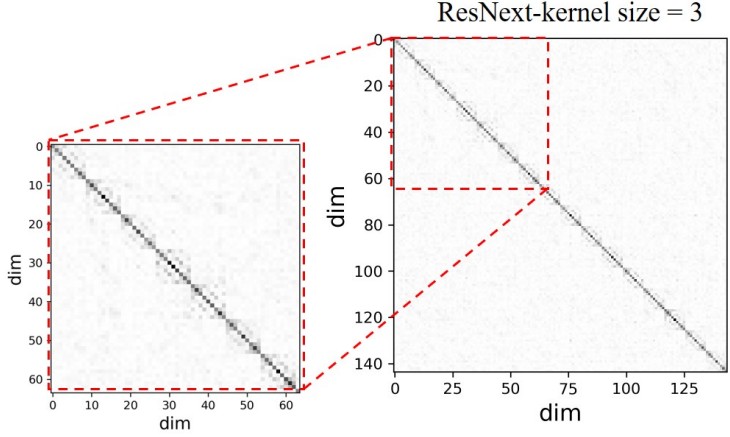

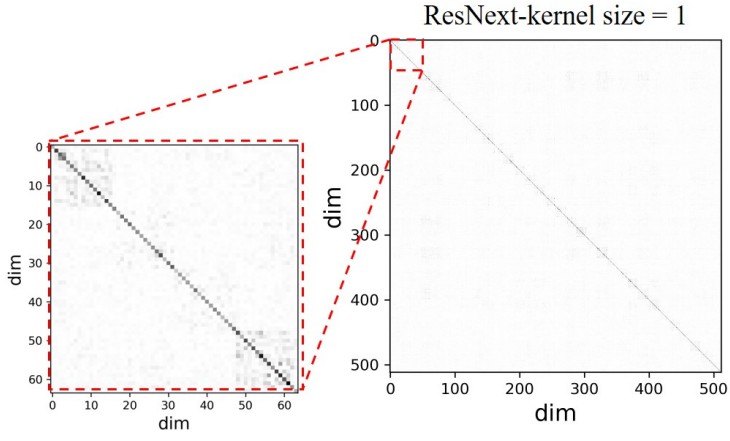

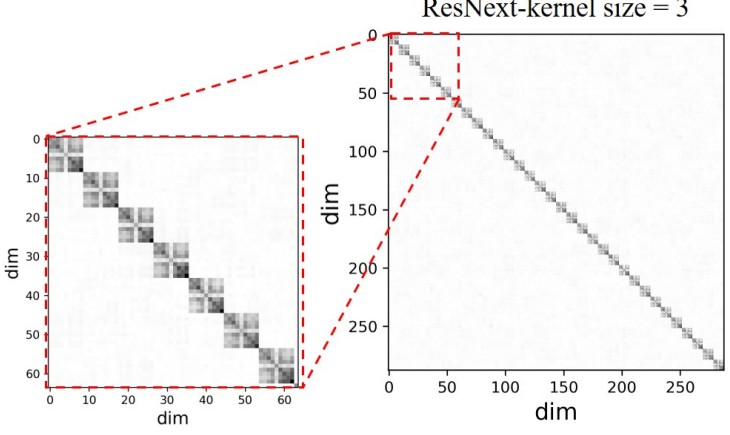

## Q.8    MOBILENET

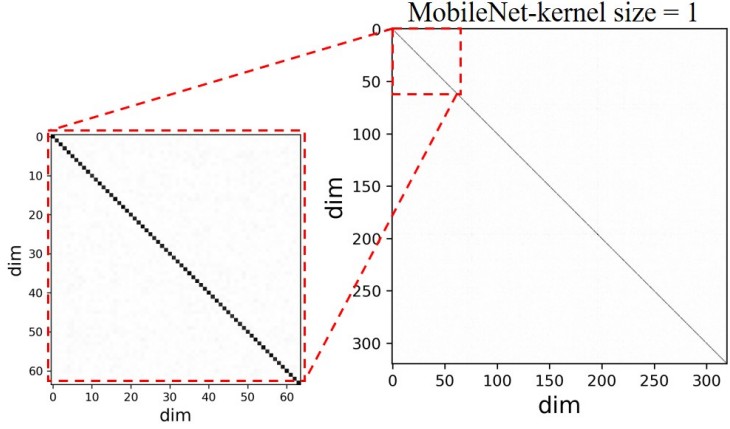

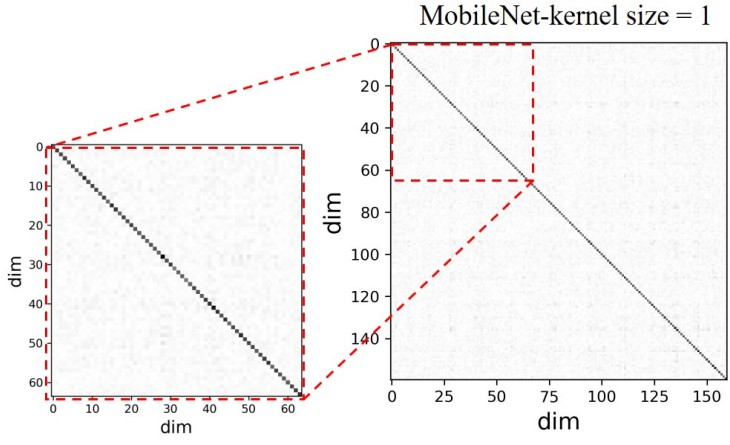

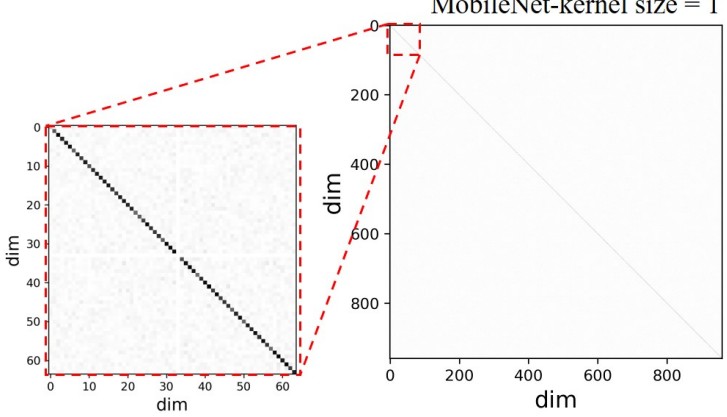

# R  DETAILS OF EXPERIMENTS

Table 64: Implementation detail for Table 6.

| Model | Dataset | Batch size | Epoch | Optimizer | schedule | wd | gamma |
|---|---|---|---|---|---|---|---|
| VGG16 | CIFAR10 | 128 | 180 | SGD(0.9) | 82/164 | 1.00E-04 | 0.1 |
| VGG16 | CIFAR100 | 128 | 180 | SGD(0.9) | 82/164 | 1.00E-04 | 0.1 |
| VGG16 | ImageNet | 256 | 120 | SGD(0.9) | 30/60/90 | 1.00E-04 | 0.1 |
| ResNet56 | CIFAR10 | 128 | 180 | SGD(0.9) | 82/164 | 1.00E-04 | 0.1 |
| ResNet56 | CIFAR100 | 128 | 180 | SGD(0.9) | 82/164 | 1.00E-04 | 0.1 |
| ResNet34 | ImageNet | 256 | 120 | SGD(0.9) | 30/60/90 | 1.00E-04 | 0.1 |

| | |
|---|---|
| Batch size | train batchsize |
| Epoch | number of total epochs to run |
| Optimizer | Optimizer |
| schedule | Decrease learning rate at these epochs |
| wd | weight decay |
| gamma | learning rate is multiplied by gamma on schedule |

There are several additional experiments of Fig. 5 in following figures. They show the Spearman's rank correlation coefficient (Sp) among $\ell_1$ pruning, $\ell_2$ pruning and **GM** pruning for all the experiments in Table 3. These experiments further visualized the strong similarities of these pruning methods in different situations.

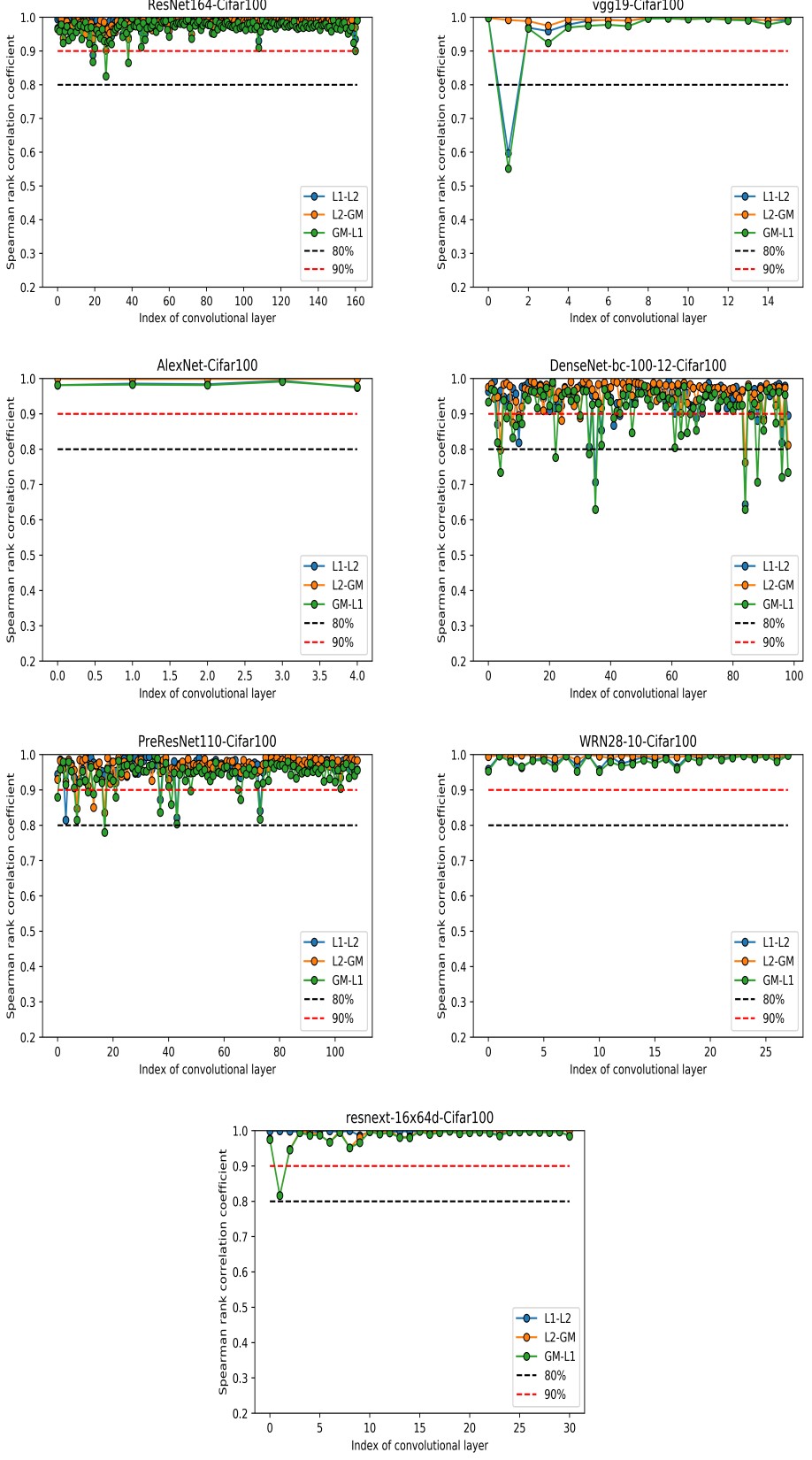

Figure 27: Network Structure

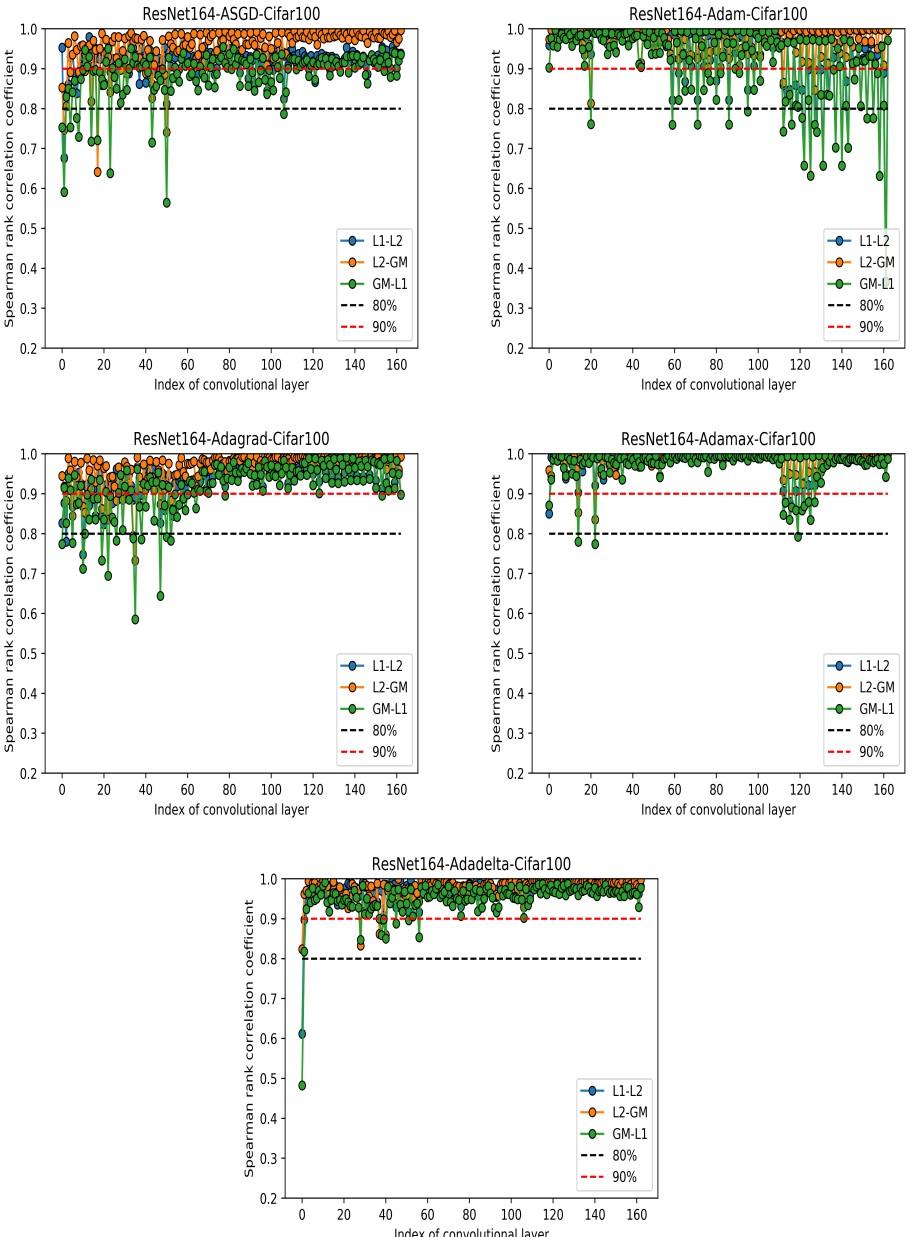

Figure 28: Optimizer

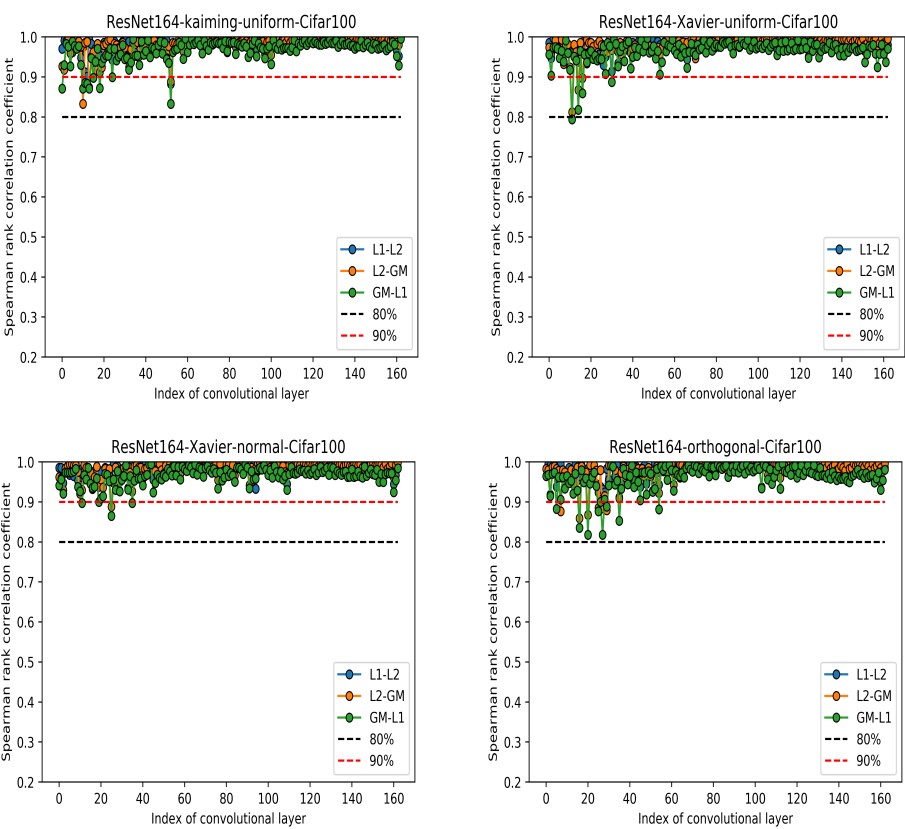

Figure 29: Initialization

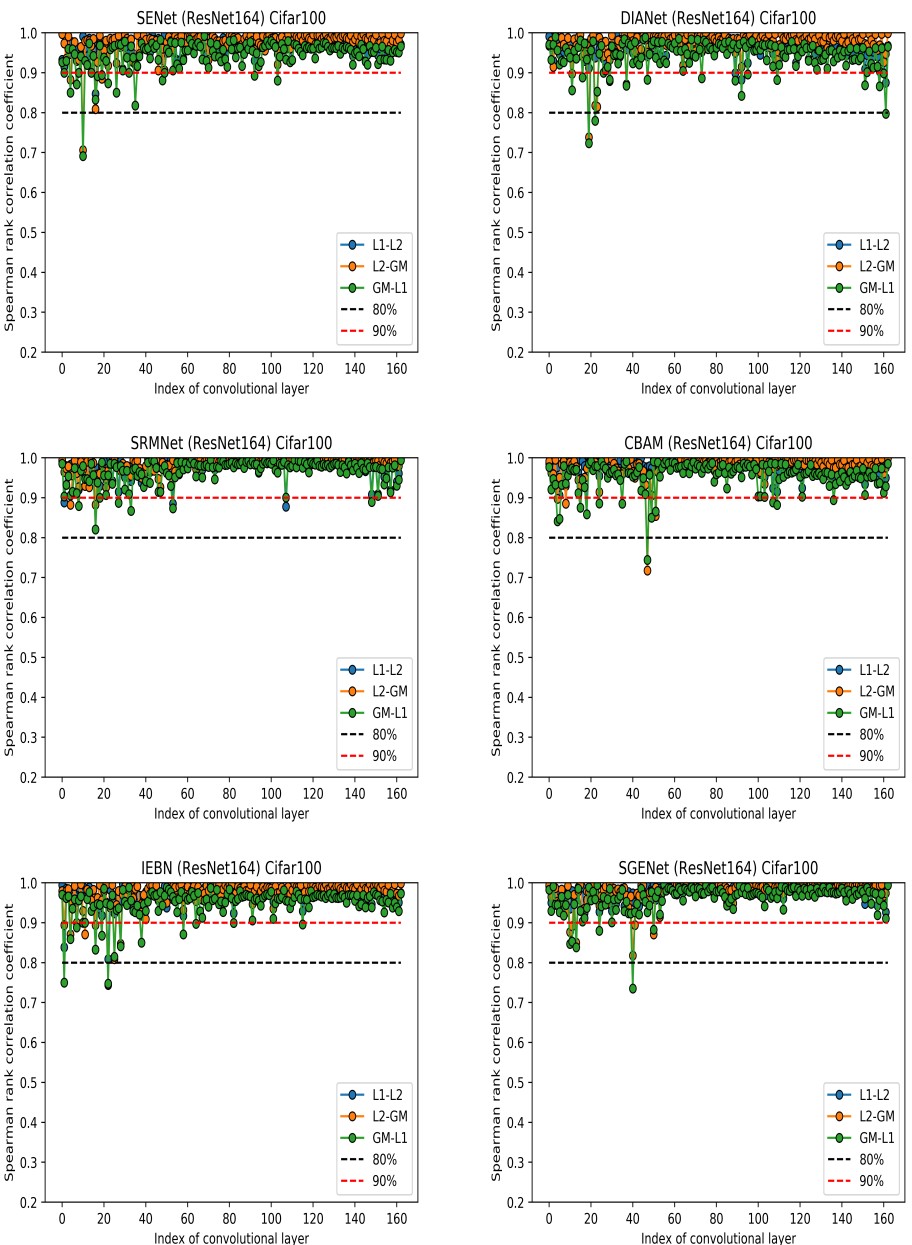

Figure 30: Attention mechanism

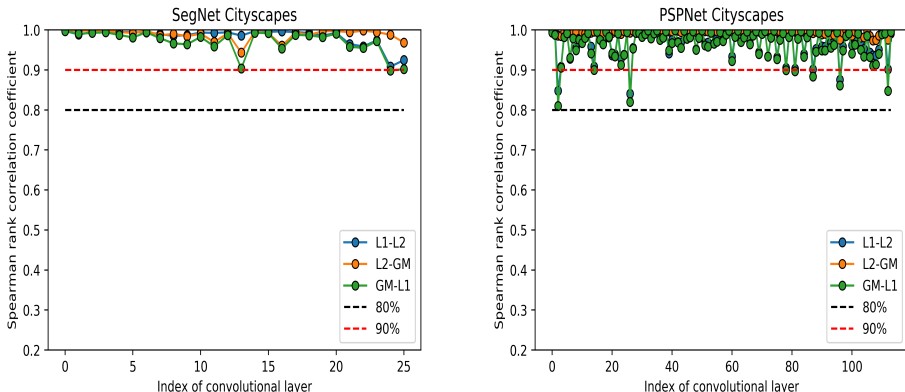

Figure 31: Other task: segmentation

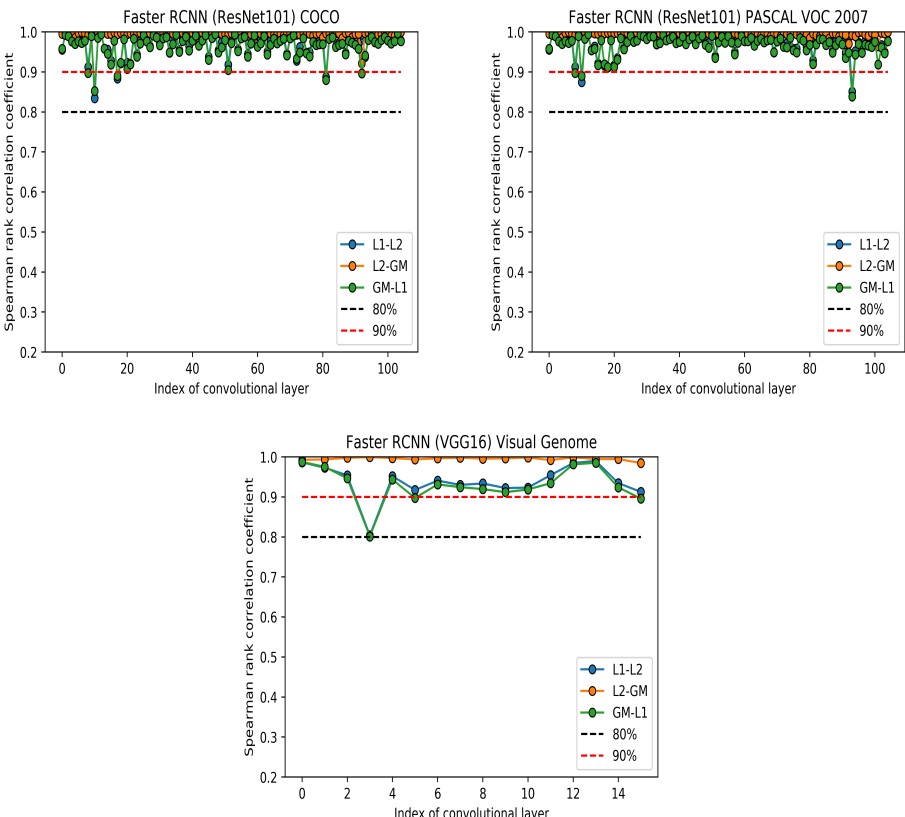

Figure 32: Other task: Faster RCNN

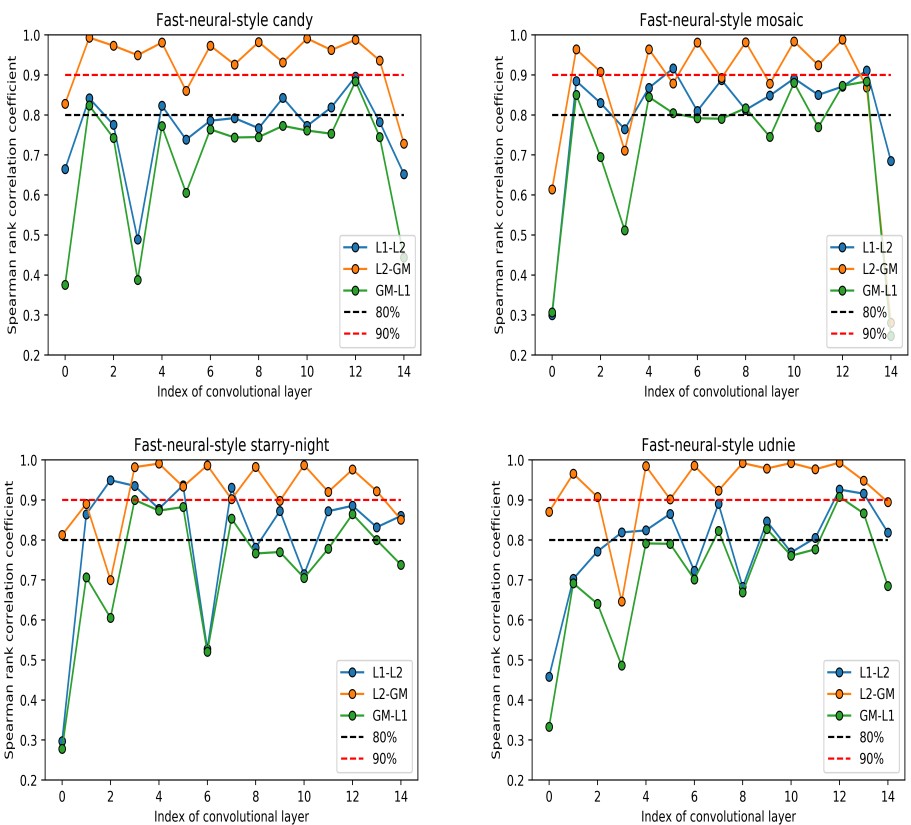

Figure 33: Other task: style transfer

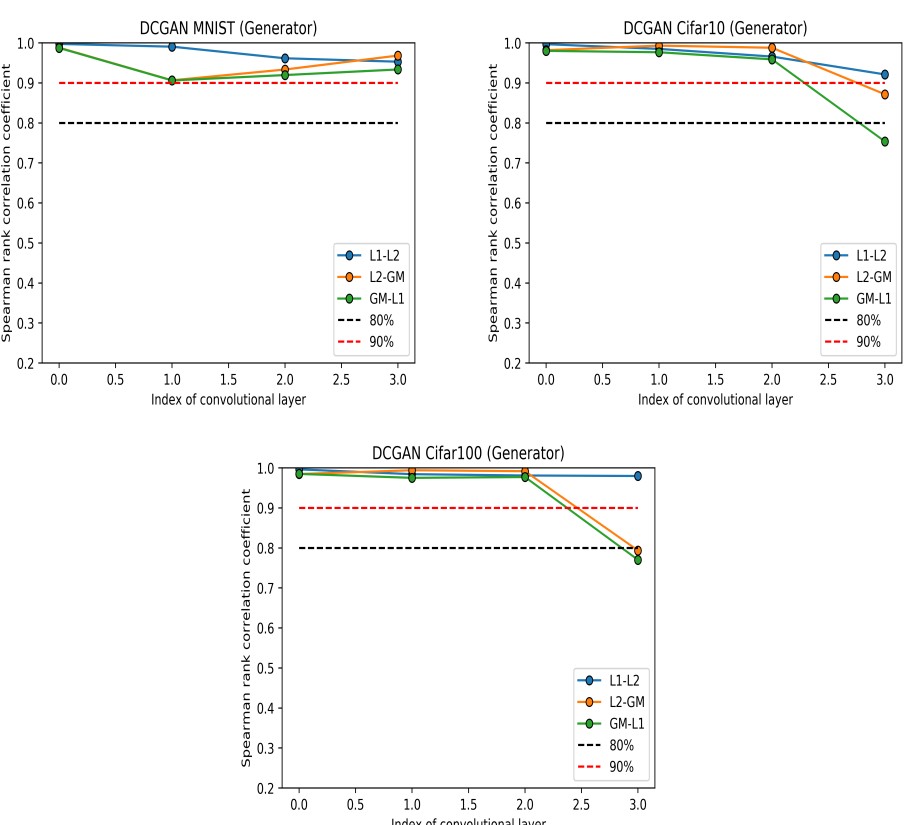

Figure 34: Other task: GAN

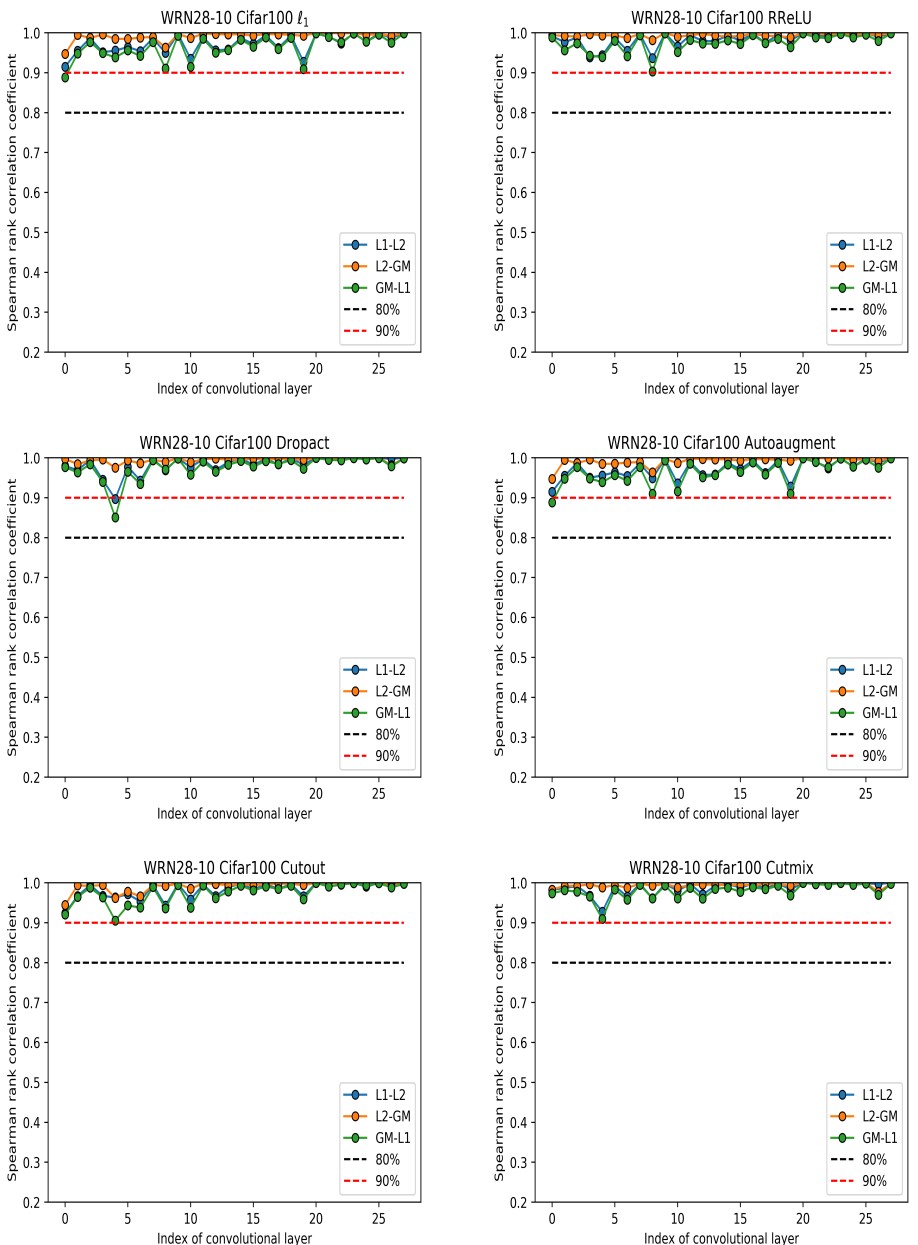

Figure 35: Other task: Regularization

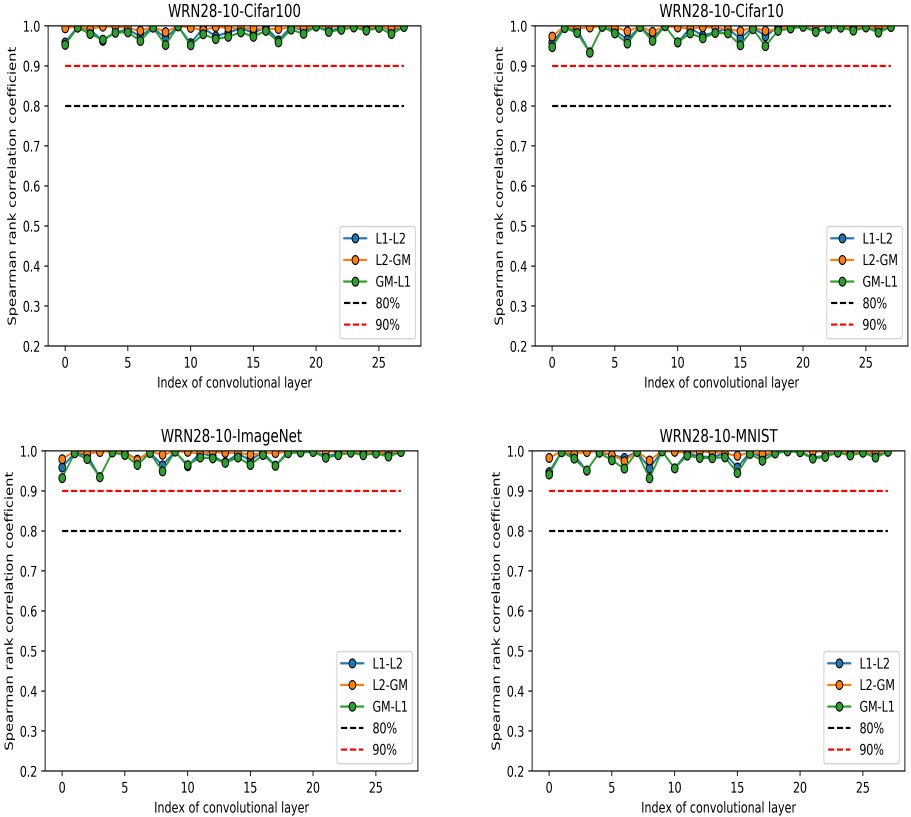

Figure 36: Dataset

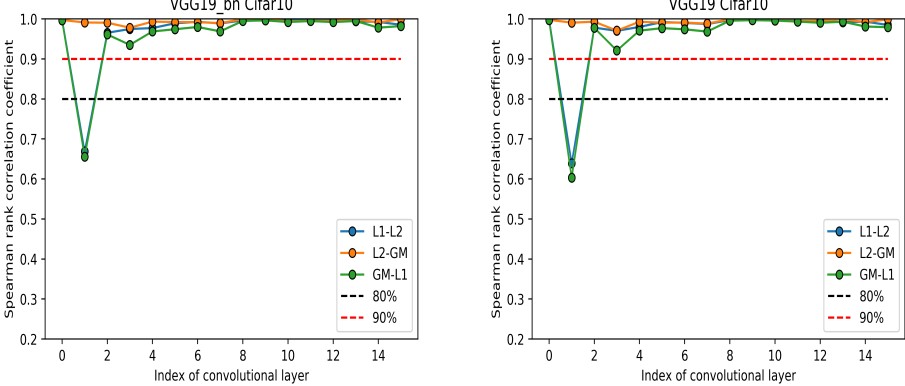

Figure 37: Batch normalization

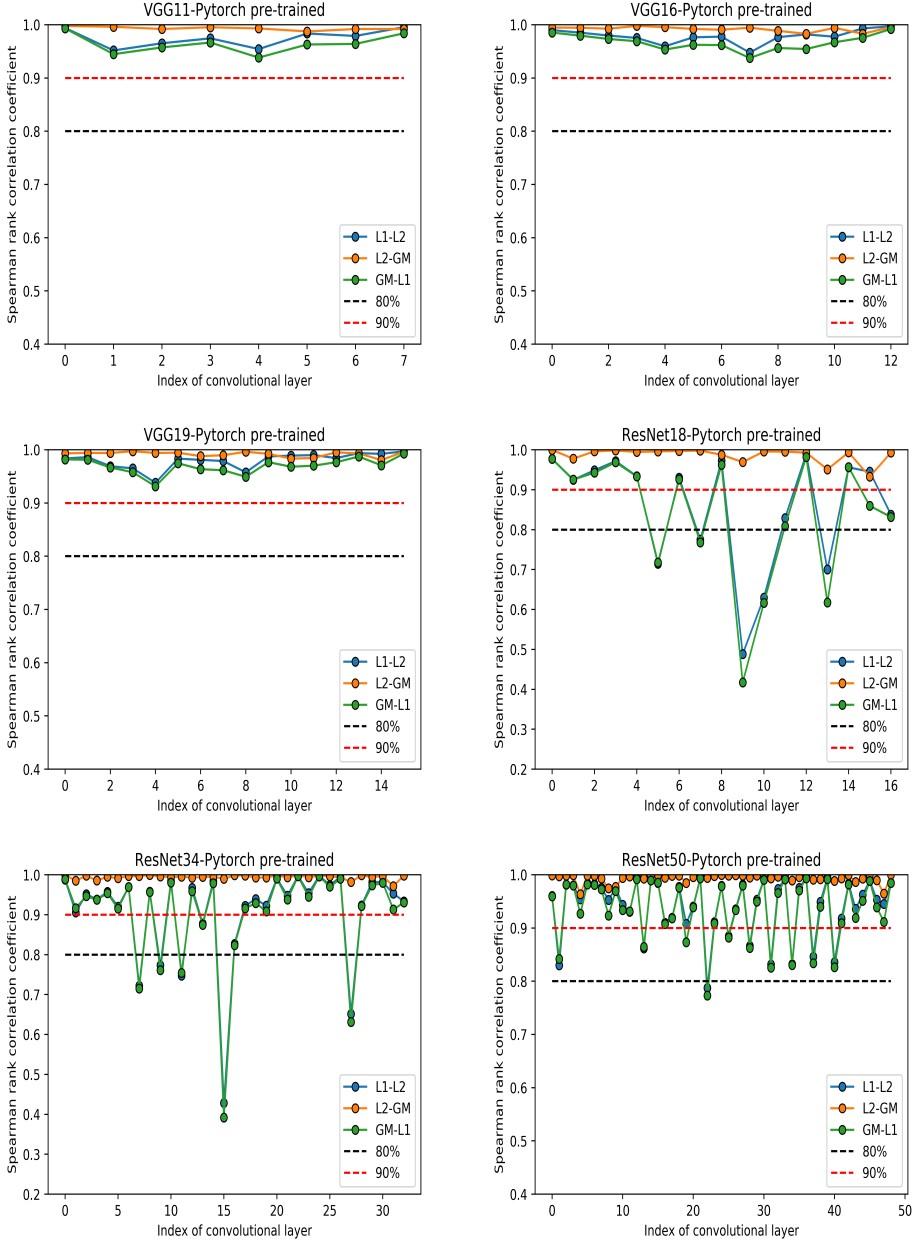

Figure 38: Pytorch pre-trained Model

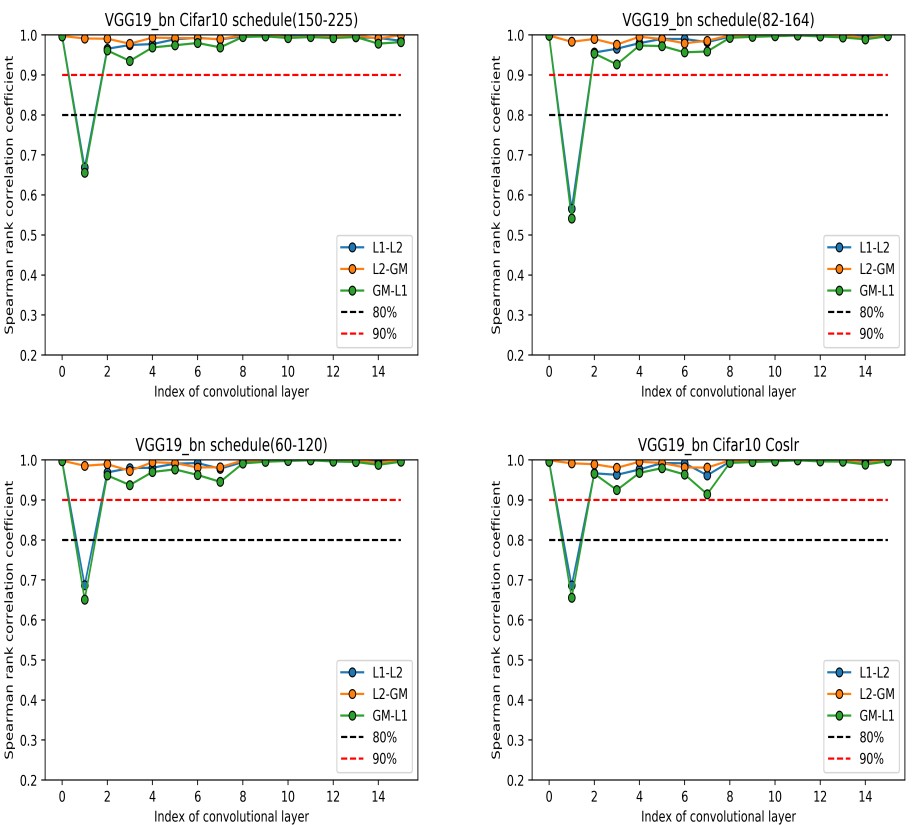

Figure 39: Learning rate

