# OpenReview forum: "Rethinking the Pruning Criteria for Convolutional Neural Network"
_ICLR.cc/2021/Conference — Reject_

### Official Review · AnonReviewer4 · 2020-10-26
**An interesting but not good enough paper**

**Rating:** 4
**Confidence:** 4

**Review:**

### About weight distribution assumption

The authors assume that all well-trained filters are i.i.d and follow a Gaussian-alike distribution, plus the co-variance matrix is a diagonal matrix. In other words, each weight within the filter follows a Gaussian distribution and is i.i.d because their co-variance equals to 0. **PS: cov(x,y) = 0 is equivalent to independence for Gaussian distribution, though the conclusion does not hold for general cases.**

However, CNN works because it takes advantage of the locality of images. In other words, these weights in filters should be highly correlated to extract features (though may not be linearly correlated), which seems to contradicte the assumption.



### About contributions of the paper

As I understand, the contributions of the paper are mainly two folded: 1) Weight distribution assumption and its correponding conclusions. 2) The paper reveals the similarity of some previous pruning algorithms and finds they may not be good criteria.

I am kind of worry about the reasonability of the first contribution (Please see last section for details). The second contribution does not seem to be sufficient for a formal paper. We may prefer some more materials. For example, why these findings are important? How do they inspire subsequent research? What should we do to avoid the problem in pruning? etc.

Totally speaking, I think the authors analyze the problem from a very interesting perspective. It will become a good paper with some more improvement, but I do not think it has been good enough for publication now.

### Updates after rebuttal

About the contributions of the paper, it does provide some interesting points of view.

The authors revise the paper and modify their basic assumption. However, it seems to be a hollow assumption without any guarantee. The concerns are as follows: First of all, why the authors claim it as a diagonal matrix when actually observing it as a block-diagonal matrix? Secondly, considering the updated assumption is only tested on one layer of VGG and ResNet, I am really worried about its generality and reproducibility.

Totally, the weight distribution assumption is the cornerstone of the paper, but the assumption seems to be not convicing. Specifically, the assumption in the first version of the paper is a paradox to some extent, and the revised version casually modifies the assumption without too much verfication. I have to decrease my rating to 4.

---

> ### Author Response · Authors · 2020-11-21
> **Response to AnonReviewer4**
>
> We thank the reviewer for the constructive feedback.
>
>
> **Q1: About weight distribution assumption:**
>
>
> A1: We strongly agree with the point that the parameters in filters should be highly correlated (though they may not be linearly correlated). In the experiments in Section2 (Fig.2), we saw that for a convolutional filter, the parameters in the same channel are linearly related, but the parameters on different channels have only very low linear correlation.
>
> We improve the description of CWDA in revision, please refer to Section 2. In revision, we argue that the convolutional filters is follow a Gaussian-alike distribution $\mathbf{N}(\mathbf{0},c^2\cdot \mathbf{\Sigma_{{block}}})$ ,where the covariance matrix $\mathbf{\Sigma_{{block}}}$ is a block diagonal matrix (the off-diagonal elements are 0).
>
> Then, we find that the block diagonal matrix can be relaxed to a diagonal matrix in the convolutional layer. The statistical tests in Section 2.1 and the estimation (as shown in Fig.10) by CWDA show the reasonability of this relaxation.
>
> Therefore, it does not contradict our assumption.
>
> **Q2: Why these findings are important? How do they inspire subsequent research? What should we do to avoid the problem in pruning? etc.**
>
>
> A2: We add experiments and analysis about the similarity and applicability for :
> 1. Global pruning. (please refer to general concern 2)
> 2. Different types of pruning criteria. (please refer to general concern 3)
>
> The importance of the findings in our paper and the suggestions for future methods can refer to general concern 4.

---

> ### Author Response · Authors · 2020-11-23
> **Thanks for your updated comments**
>
> Thank you again for your comments.
>
>
>
> **Q1: Why the authors claim it as a diagonal matrix when actually observing it as a block-diagonal matrix?**
>
>
> A1:  For a convolutional filter, the fact that the covariance matrix $\mathbf{\Sigma_{\text{block}}} \in \mathbb{R}^{(k^2\times N_{i+1})\times (k^2\times N_{i+1})}$ can be regarded as a diagonal matrix, which has explained in Section 2.
>
>
>
> Specifically, since the kernel size $k$ is a small constant (like 1 or 3) and $N_i \gg 1$, the length $k^2$ of each block is much smaller than the length $k^2\cdot N_{i}$ of the covariance matrix in most convolutional layers. Therefore, the block diagonal matrix $\mathbf{\Sigma}_{\text{block}}$ can be relaxed to a diagonal matrix, as shown in Fig.2 (b) and (d).  A lot of experiments and analysis in our paper demonstrate the reasonability of this relaxation, *e.g.*, the estimation using CWDA (relaxed) almost coincides with the actual results obtained from the real networks (Fig. 10). Furthermore, for the sake of rigor, we have discussed *why CWDA sometimes does not hold*  in Section 5.1.
>
>
>
>
>
>
> **Q2: Secondly, considering the updated assumption is only tested on one layer of VGG and ResNet, I am really worried about its generality and reproducibility.**
>
> A2: We show more experiments to verify the observations for the updated assumption in Appendix Q [page 117]. **For generality**, we have already tested VGG16/VGG19/ResNet18/ResNet50/AlexNet/DenseNet/ResNext/MobileNet trained on ImageNet in our experiments. And the experiments show that these covariance matrices tend to be the block-diagonal matrices, which are consistent with the observation in Fig. 2; **For reproducibility**, all the networks in these experiments are from [ *torchvision models zoo*](https://pytorch.org/docs/stable/torchvision/models.html).

---

### Official Review · AnonReviewer3 · 2020-10-28
**Interesting insights into norm-based filter pruning but it lacks a broader discussion**

**Rating:** 5
**Confidence:** 4

**Review:**

## Summary

The paper discusses various baseline scoring mechanisms used for filter pruning that are norm-based and finds that none of the scoring mechanisms are particularly effective at pruning filters from CNNs. Moreover, all methods seem to perform very similar to each other. These conclusions are based on a theoretical (and experimental) analysis of the various scoring mechanisms under the assumption that trained filter weights follow a Gaussian-like distribution, which reveals that in this case the scoring mechanisms are insufficient to reliably discern the importance of filters since the resulting scores are very similar. The Gaussian-like assumption, which is termed "Convolution Weight Distribution Assumption" (CWDA), is validated through a large range of experiments on different architectures and data sets.

## Score

I enjoyed reading the paper and I found the resulting conclusions really interesting since it sheds light on some of the mechanisms behind filter pruning methods, which are not as well investigated as mechanism behind unstructured (weight) pruning.

Nonetheless, I think there a is couple of things that the authors should consider before the paper is ready for publication, namely the inclusion of more advanced filter pruning techniques [1, 2, 3] in the discussion and also a set of recommendations for the community that could help design more informative scoring mechanisms for filter pruning in future work.

More details are provided below.

## Ways to Improve My Score

To improve the quality of the paper, I believe the authors should include the following two aspects:
1. Additional filter pruning methods: There has been significant progress in recent years in developing more advanced filter pruning techniques, such as [1, 2, 3] including techniques that are data-dependent, i.e., consider the layer activation in computing the filter score. Do these methods suffer from the same inability to discern the importance of filters? Or are these methods already a stepping stone in the right direction towards building more meaningful pruning criteria? Having the baseline methods alone is not sufficient in order for the community to build on top of this work in a meaningful way.
2. Recommendations for improved filter pruning methods in the future: As it stands now, the paper provides little guidance what future research into filter pruning should consider when designing novel pruning methods. I think this would be crucial though in order for the community to appreciate the paper. This is also linked to the first point I mentioned since some of the existing advanced filter pruning methods potentially already contain at least partial solutions to the problem presented here and could thus guide future research as well.

I mention additional feedback that affects my score in the "Weaknesses" section. The "minor feedback" contains feedback that did not significantly affect my score.

## Strengths

* I think the conclusions are very interesting and confirm my own experience with filter pruning. A lot of the baseline approaches (i.e. the norm-based approaches) seem to work more or less the same, especially in a standardized prune+retrain setting.

* The CWDA assumption is verified using a large set of experiments and the authors also discuss the limitations of the assumptions. This is really encouraging to see and I appreciate the rigor.

* The theoretical analysis based on the CWDA assumption seems technically sound and is also back up by appropriate experiments. Figure 1 is really interesting, in particular, as it shows how small the variance of norm distribution is.

## Weaknesses

* The lack of additional filter pruning methods as stated above. The inclusion of those is crucial I believe as it may lead to some really interesting additional conclusions.

* The authors discuss the limitations when CWDA does not hold. I wonder if they could dig a little deeper to see whether in those cases norm-based filter pruning is more effective at identifying important filters.

* The writing in parts is somewhat confusing and as a results it's sometimes hard to discern the meaning. I believe the paper would benefit from an additional revision to ensure that it is accessible to a broader audience.

## Other Minor Feedback

* Typos:
  * in (2) of the contribution list: "...when the network exists too many filters."""
  * Page 7, bottom: "..., which shows that the GM method also exist the same problem ..."
  * Page 8, top: "In CVPR 2019 Oral, ...."
  * Page 8, Section 5.1, (1): "...converges to a bad local minimal ..."
  * Page 8, Section 5.1, (2): ...are almost the layers that are ..."
  * Please check for other typos as well.

* Is there a difference between $F_{i,j}$ and $F_{ij}$? I saw both notations in the paper and wasn't sure.

* I didn't carefully check all the math but it seems to be reasonable. I think it would help potential readers to include high-level summaries of each theorem in the main part of the paper. That way the paper could be made more accessible to a larger audience.

* Maybe it would be more useful to include the results of the statistical tests as part of a code repo that will be released upon publication. It feels quite difficult and slow to navigate many pages of tabulated results.

 * Maybe sort the indices in Table 8 differently. Then it might be easier to see the overlap between the chosen filters for each method.

 * What about fully-connected layers? Could the authors observe something similar in those cases? Note that "filter pruning" can also be applied to fully-connected layers where neurons corresponds to filters.

## References

1. [ThiNet: A Filter Level Pruning Method for Deep Neural Network Compression](http://openaccess.thecvf.com/content_iccv_2017/html/Luo_ThiNet_A_Filter_ICCV_2017_paper.html)
2. [Provable Filter Pruning for Efficient Neural Networks](https://openreview.net/forum?id=BJxkOlSYDH)
3. [Good Subnetworks Provably Exist: Pruning via Greedy Forward Selection](https://proceedings.icml.cc/paper/2020/hash/8b6a80c3cf2cbd5f967063618dc54f39-Abstract.html)

---

> ### Author Response · Authors · 2020-11-21
> **Response to AnonReviewer3**
>
> We thank the reviewer for the constructive feedback.
>
> **Q1: About the empirical results for other pruning criteria. (Norm-based, Importance-based, BN-based and Activation-based)**
>
>
> A1:Please refer to the General Concern 3.
>
> **Q2: Recommendations for improved filter pruning methods in the future.**
>
>
> A2: Please refer to the General Concern 4.
>
> **Q3: The lack of additional filter pruning methods as stated above. The inclusion of those is crucial I believe as it may lead to some really interesting additional conclusions.**
>
>
> A3: About the empirical results for other pruning criteria. Please refer to General Concern 3. There are several interesting conclusions, *e.g.*, the similarity and applicability problems are not only for Norm-based criteria but also for some other types of criteria. Additionally, the related discussion of global pruning is also shown in General Concern 2.
>
>
> **Q4: The authors discuss the limitations when CWDA does not hold. I wonder if they could dig a little deeper to see whether in those cases Norm-based filter pruning is more effective at identifying important filters**
>
>
> A4: Thank you for your advice. In Section 5.1, we show that there are several situations where CWDA does not hold, and in these cases, the Norm-based filter pruning methods have the following findings:
> 1. The network is trained not good enough. We use the following two settings to train WRN in Appendix A (Fig.12) [page 14]:
> (1) Without data augmentation; (2) Uniform Initialization + 6 epochs training. We find that it is similar to the results in Fig.1, $\ell_2$ still can not identify important filters in these settings.
>
> 2. The number of filters is insufficient, or the dimensions of the filter are not large enough. In fact, the first few convolutional layers shown in Fig. 8 satisfy these situations. It shows that the $\ell_2$ criterion does not have an applicability problem in these layers.
>
>
> Therefore, when CWDA does not hold, it is not easy to draw a conclusion on the applicability, i.e., whether the Norm-based criteria are more effective at identifying important filters.
>
>
>
> **Q5: About writing.**
>
>
> A5: We thank the reviewer for his valuable comments and we revised our paper in revision. Also, we include high-level summaries of each theorem (original paper) in the main part of the paper in Theorem1 (revision).
>
>
> **Q6: Is there a difference between $F_{i,j}$ and $F_{ij}$?**
>
>
> A6: This is a typo and we updated in revision. $F_{ij}$ is correct.
>
>
> **Q7: About pruning for fully-connected layers (FC).**
>
>
> A7: Please refer to General Concern 6.

---

### Official Review · AnonReviewer1 · 2020-10-29
**Experiments to validate two limitations of magnitude based pruning methods**

**Rating:** 3
**Confidence:** 5

**Review:**

This paper analyzes the current limitations of existing magnitude-based pruning methods. First, the paper focuses on the similarities between three methods and then focuses on the redundancy in large networks. The paper also analyzes the weight distribution for a well-trained network and propose CDWA as a way to prove this distribution.  My main concern with this paper is the contribution to the field.

Two main comments:
1. If I understand correctly, the paper focuses on the distribution at the parameter level and shows it follows a Gaussian Distribution. If that is the case, that is a well-known result especially when the network is trained with a weight decay to regularize.

Then, the paper shows the similarities between three magnitude based methods (L1, L2 and GM). I guess I am not very surprised the first two are similar in the structure and slightly different in performance as that is expected. The rank correlation is interesting tho. Would be more interesting to show the same comparison with importance based methods or other methods not using the magnitude only.

2. In terms of redundancy, I am not sure I understand the point there. It is also known that training of neural networks leads to redundancy and that is the main reason for some particular postprocessing or compression algorithms based on that (see Exploiting Linear Structure Within Convolutional Networks for Efficient Evaluation -- Denton et al; or Compression aware training of Neural Networks -- Alvarez and Salzmann; and some others in the literature). Magnitude based approaches do not consider the interaction between parameters and therefore still lead to redundancy. Methods aiming at reducing redundancy usually do not provide good compression ratios as the architecture is harder to prune.

The paper also suggests l2 is regarded as importance. I think that should be reconsidered as L2 is the magnitude and there are many works using other criteria referred as importance (for instance gradient *magnitude).

All in all, not sure the real contribution to the community.

---

> ### Author Response · Authors · 2020-11-21
> **Response to AnonReviewer1**
>
> We thank the reviewer for the constructive feedback.
>
> **Q1: If I understand correctly, the paper focuses on the distribution at the parameter level and shows it follows a Gaussian Distribution. If that is the case, that is a well-known result especially when the network is trained with a weight decay to regularize.**
>
>
> A1: Please refer to Appendix O [page 37] in the revision. We update the KS test under different weight decay (wd=0, 1e-3, 3e-4). From these results, weight decay can make the parameters have a higher p-value in the KS test, but when wd=0, the p-value in the KS test is also much greater than 0.05. Therefore, we argue that wd contributes to normality, but it is not necessarily why the parameters follow the Gaussian-alike distribution.
>
>
> **Q2: the paper shows the similarities between three magnitude based methods ($\ell_1$, $\ell_2$ and $\mathbf{GM}$). I guess I am not very surprised the first two are similar in the structure and slightly different in performance as that is expected.**
>
>
> A2: Intuitively, $\ell_1$, and $\ell_2$ should indeed be very similar, but some results in our paper may be counter-intuitive:
>
>
> 1. As the analysis in Section 3. the similarity between $\ell_1$ and $\ell_2$ is not always so strong (Fig. 5), especially in the transitional area.
> 2. From the proof of Theorem 1 (in revision), comparing ($\ell_2$, $\mathbf{Fermat}$) and ($\mathbf{Fermat}$,$\mathbf{GM}$), the upper bound $B(n) = \frac{1}{n}$ of ($\ell_1$, $\ell_2$) is the loosest.
> 3. In Section 5.3 (Fig.9), the results of global pruning show that the similarity between $\ell_1$ and $\ell_2$ is not as strong as the one in layer-wise pruning. For this phenomenon, we also explain it according to CWDA in Appendix A [page 14].
> 4. In Section 5.2 (Fig. 8), the experiments show that $\ell_2$ has applicability problems, but $\ell_1$ does not if the network has a large number of convolutional filters. In Section 4, CWDA also explains the phenomenon.
>
> **Q3: About the empirical results for other pruning criteria. (Norm-based, Importance-based, BN-based and Activation-based)**
>
>
> A3:Please refer to the General Concern 3.
>
>
> **Q4:  About redundancy.**
>
>
> A4: Yes, I agree with you, and this may be the reason why the performance of Norm-based methods is relatively weak. However, Norm-based pruning criteria are still the representative criteria and are widely used in industrial applications. To further investigate them, in this paper, we include the discussion of similarity and applicability problems. According to the smaller-norm-less-informative assumption[1], redundancy refers to the filter with a small importance score measured by the Norm-based criterion.
>
>
> **Q5: The paper also suggests $\ell_2$ is regarded as importance. I think that should be reconsidered as $\ell_2$is the magnitude and there are many works using other criteria referred as importance (for instance gradient magnitude).**
>
>
> A5: We agree that $\ell_2$ should be regarded as the magnitude. However, a lot of previous works [2][3][4][5][6][7] consider $\ell_2$ as the importance score to prune the convolutional filters. Moreover, we also update the experiments about other types of pruning criteria (e.g., Activation-based, Importance-based, BN-based, Norm-based). Please refer to the General Concern 3, where the Importance-based criteria are the gradient magnitude criteria mentioned in the review.
>
>
> [1] Ye J, Lu X, LinZ, et al. Rethinking the smaller-norm-less-informative assumption in channelpruning of convolution layers[C]. ICLR 2018.
>
>
> [2] Frankle J, Carbin M. Thelottery ticket hypothesis: Finding sparse, trainable neural networks[C]. ICLR,2019.
>
>
> [3] Renda A, Frankle J, CarbinM. Comparing rewinding and fine-tuning in neural network pruning[C]. ICLR,2020.
>
> [4] He Y, Kang G, Dong X, etal. Soft filter pruning for accelerating deep convolutional neuralnetworks[C]//IJCAI. 2018.
>
>
> [5] He Y, Ding Y, Liu P, etal. Learning Filter Pruning Criteria for Deep Convolutional Neural NetworksAcceleration[C]//CVPR 2020.
>
>
> [6] Li B, Wu B, Su J, et al.Eagleeye: Fast sub-net evaluation for efficient neural network pruning[C]//ECCV,2020.
>
>
> [7] Li H, Kadav A, DurdanovicI, et al. Pruning filters for efficient convnets[C]. ICLR 2017.

---

### Official Review · AnonReviewer2 · 2020-10-29
**Raises an important aspect, but there are still some reservations**

**Rating:** 5
**Confidence:** 4

**Review:**

**Summary:** The goal of the paper is to bring into attention that many norm-based pruning criteria used for structured pruning are very similar, in that their ranking of the redundant filters is highly correlated. The key ingredient is the CWDA assumption that filters in a particular convolutional layer are iid and approximately follow a Gaussian distribution, which is shown based on extensive statistical hypothesis testing. Based on this assumption, they prove that these pruning criteria are roughly the same.

**Pros:**
I like the nature of the question asked in this paper. These are fundamental questions, like how does the distribution of weights in a particular layer evolve and look like, etc, which do not get as much attention. So in that sense, this paper will be important for the pruning community (where there is a deluge of papers proposing a new criterion) to be aware of, and understanding this aspect they would be able to design a pruning criterion that actually matters. The paper is backed with extensive results testing their CWDA assumption and proving how the criterion are overlapping in their functionality.


**Cons:**
There is an over-selling of claims, as this "rethinking" of criteria is only for norm-based methods, and there are lot of other established methods as well. Then this all holds for layers which are wide enough, but in the end for pruning these layers would not be as sensitive either (and layers like input, last fully-connected, or where there is a transition).
Also, at some other places like in the results of different pruning criteria, the experimental methodology and presentation is rather loose. I would have also liked to see at least the empirical results for global pruning because there even some out of these norm criteria might be better than others.

**Detailed comments**:

*Similarity between pruning criteria*:

-First of all, Molchanov et al 2019 https://openaccess.thecvf.com/content_CVPR_2019/papers/Molchanov_Importance_Estimation_for_Neural_Network_Pruning_CVPR_2019_paper.pdf does look at the spearman rank correlations, so I guess you should also mention this in the context of "blind-spot".

-Next the criteria that you compare are only based on (absolute/relative) norm of the filters. And, I understand that your main point is showing that many norm based methods are ranking filters in more or less the same way. However, there are several other established and widely used measures (such as Network slimming: Liu et al 2017; EigenDamage: Wang et al 2019). Hence, using the title “Rethinking the pruning criteria…” is too broad of a claim, and should be qualified to include ”norm-based”.

-Also, the paper could have been made stronger by comparing these other measures, empirically at the least.

*Empirical analysis*:
- Table 4: It is not clear what is the pruning level here?  If you are pruning to small levels, then it does not matter as much of course (also known from results of other papers).  Here, all the methods recover performance after fine-tuning, which shows that this is not yet a good-test bed. The testing should be done precisely in the regime of high pruning ratios (> 90%, 95%), so that you actually see the difference. Can you perhaps provide results in such settings?

- Further, can you share some correlation numbers for global pruning? I know that your focus is layer-wise pruning, which is perfectly fine. However, global pruning methods outperform layer-wise methods (at least this is known for unstructured pruning), and my intuition is that it might also hold for structured pruning (also, global methods would be better from the large minimum norm requirement of He et al 2019).

- Table 1: I do not see much worth of showing that top 8 filters (out of > 100s) are similar for all criterions, as the few most redundant filters will be identified almost equally well by any sensible criterion.

*Other important questions:*
How do you choose \sigma_0 in testing? What is the smallest value that you can set, such that most of the tests pass?
"Need to be trained well enough": Why is this a concern?

*Related work discussion*:
-Pruning: References go to as back as LeCun et al 1990, Hassibi & Stork 1994, etc. Otherwise it gives an impression that this whole area has just started.

-"one-shot pruning”: The definition of one-shot does not include fine-tuning. I think saying, "one-shot pruning + fine-tuning”, is more accurate

*On writing:*
- There are actually quite a few typos, but still not too bad yet something not be proud of. Towards the last couple of pages, there are many more typos.
"about the pruning criteria as follow”;
"when the network exists too many filters” -> "when the network has too many filters;
“First, two theorems are shown” -> “First, we show the following two theorems"

- At many places, you have subscripts instead of superscript: "In ith layer”

- "In CVPR 2019 Oral”: I don’t think this is a good way of writing or emphasising a paper. Otherwise, we will have things like, “The best paper award winner XYZ et al says, ….”, “The paper XYZ et al, which features a Nobel prize winner,  ….”. I hope you get the point.

**Score**: Right now, owing to the above points, I have some reservations. If you can answer these questions better, I would happily increase my score towards acceptance.

---

> ### Author Response · Authors · 2020-11-21
> **Response to AnonReviewer2**
>
> We thank the reviewer for the constructive feedback.
>
>
> **Q1: There is an over-selling of claims, as this "rethinking" of criteria is only for Norm-based methods.**
>
>
> A1: Please refer to the General Concern 3.
>
>
> **Q2: Then this all holds for layers which are wide enough, but in the end for pruning these layers would not be as sensitive either (and layers like input, last fully-connected, or where there is a transition). Also, at some other places like in the results of different pruning criteria, the experimental methodology and presentation is rather loose.**
>
>
> A2: The experiments on the general layers (not specific for the wide-enough layers) are shown in Fig.7 (similarity) and Fig.8 (applicability). The experiment of last fully-connected layer are shown in General concern 6. We update the experimental methodology and presentation part in revision.
>
>
> **Q3: About the empirical results for global pruning.**
>
>
> A3: Please refer to the General Concern 2.
>
>
> **Q4: First of all, Molchanov et al 2019 https://openaccess.thecvf.com/content_CVPR_2019/papers/Molchanov_Importance_Estimation_for_Neural_Network_Pruning_CVPR_2019_paper.pdf it does look at the spearman rank correlations, so I guess you should also mention this in the context of "blind-spot".**
>
>
> A4: Thank you for your suggestion and we update the description in revision (Section 1).
>
>
> **Q5: About the empirical results for other pruning criteria (Norm-based, Importance-based, BN-based and Activation-based).**
>
>
> A5: Please refer to the General Concern 3.
>
>
> **Q6: Table 4: It is not clear what is the pruning level here? The testing should be done precisely in the regime of high pruning ratios (> 90%, 95%), so that you actually see the difference. Can you perhaps provide results in such settings?**
>
>
> A6: We add some experiments under different pruning ratios in Section 3 (Fig.4). In addition, we move the original Table 4 to Appendix N (Table 6) [page 36] and add experimental details. These experiments show the similarities between these pruning criteria from the perspective of test accuracy (after pruning and fine-tuning).
>
>
> **Q7: Further, can you share some correlation numbers for global pruning?**
>
>
> A7: Please refer to the General Concern 2.
>
>
>
> **Q8: Table 1: I do not see much worth of showing that top 8 filters (out of > 100s) are similar for all criteria, as the few most redundant filters will be identified almost equally well by any sensible criterion.**
>
>
>
> A8: The primary purpose of Table 1 is to show that the importance rankings of these convolutional filters measured by different pruning criteria are very similar. Moreover, we respectfully disagree with the view that “the few most redundant filters will be identified almost equally well by any sensible criterion.” In our revision, Fig.7(a) and Fig.7(d) show the Sp between most different pruning criteria in these layers are not large, which indicates that these criteria have great differences in the redundancy measurement of convolutional filters. This may lead to a phenomenon that one criterion considers a convolutional filter to be important, while another considers it redundant. We find a specific example which is shown in Appendix E [page 19].
>
> **Q9: Other important questions: How do you choose \sigma_0 in testing? What is the smallest value that you can set, such that most of the tests pass?**
>
>
> A9: In Appendix P [page 38], we choose $\sigma_0^2=0.0001$ for statistical tests. In Fig. 25, we show the passing rate under different $\sigma_0$. These experiments show that $\sigma_0$ can actually be smaller (like $\sigma_0 = 0.005$).
>
> **Q10: "Need to be trained well enough": Why is this a concern?**
>
>
> A10: We have updated the description of "Need to be trained well enough". Please refer to Section 5.1(1) in revision.
>
> **Q11: About writing.**
>
>
> A11: We thank the reviewer for his valuable comments and we have revised our paper.

---

### Author Response · Authors · 2020-11-21
**Revised instruction**

Addressing the reviewers' suggestions and concerns, we added further experimental results and made significant clarifications to the paper while striving to keep the broad ideas and findings intact. In summary, we have made the following changes:

***More Details***

We've improved the clarity of the paper:
- We improve the description of CWDA in Section 2 [page 3].
- We improve the description of the Applicability problem in Section 4 [page 6]
- We improve the description of the conditions that CWDA does not hold in Section 5 [page 7]
- We include high-level summaries of each theorem (original paper) in the main part of the paper in Theorem1 (revision) [page 6]
- We improve the details of the statistical tests in Appendix P. [page 38]

***More Experimental Results***

We've included additional results of experiments suggested by all reviewers:
- Fig.4: Test accuracy while using different pruning ratio. [page 5]
- Section 5.2: the experiments for the Similarity and Applicability problem of different types of pruning criteria (e.g., Norm-based, BN-based, Importance-based and Activation-based). [page 7]
- Section 5.3: the experiments for the Similarity and Applicability problem while using global pruning. [page 9]
- Appendix A: Further study for "Is normal distribution necessary for CWDA". [page 14]
- Appendix O: The experiments about using the different settings of weight decay. [page 37]
- Appendix Q: The additional experiments for the observation in Fig. 2. [page 117]

As suggested, we fixed the typo and writing issues of the full paper.

---

### Author Response · Authors · 2020-11-21
**General Concerns**

1. **About the title of our paper:**  As shown in the revised instruction, we add experiments and analyses on the similarity and applicability of different types of pruning criteria, as well as global pruning. We think we can keep the original title after revising.


2. **About global pruning:** We add the experiments about global pruning in Section 5.3. According to our analysis with CWDA, some Norm-based criteria using the global pruning strategy have the problem about the inconsistent magnitude of *importance* between different layers. It explains the phenomenon that these pruning criteria using the global pruning strategy sometimes cut off the network. [page 9]


3. **About other pruning criteria:** We add the experiments on the similarity and applicability for different types of pruning criteria (Norm-based, BN-based, Importance-based, and Activation-based) in Section 4. [page 6]



4. **The contributions to the community:**
    + When a new pruning criterion is proposed, it is necessary to check whether this criterion has Similarity and Applicability problems, as mentioned in our paper. We suggest that researchers need to consider more when proposing a new pruning method.
    + CWDA is not only used for channel pruning, but may also become a theoretical analysis tool in other applications of convolutional neural networks.
    + According to our analysis with CWDA, some Norm-based criteria using the global pruning strategy have the problem about the inconsistent magnitude of *importance* between different layers. It explains the phenomenon that these pruning criteria using the global pruning strategy sometimes cut off the network. To solve the problem of inconsistent magnitude, we suggest that some normalization methods should be implemented or a protection mechanism should be established, *e.g.*, to ensure that each layer has at least a certain number of convolutional filters that will not be pruned.


5. About CWDA. We improve the description of CWDA in revision. Please refer to Section 2. In revision, we argue that the convolutional filters follow a Gaussian-alike distribution $\mathbf{N}(\mathbf{0}, c^2\cdot \mathbf{\Sigma_{{block}}})$, where the covariance $\mathbf{\Sigma_{{block}}}$ is a block diagonal matrix (the off-diagonal elements are 0). Then, we find that the block diagonal matrix can be relaxed to a diagonal matrix in convolutional layer. In addition, the statistical tests in Section 2.1 and the estimation (as shown in Fig.10) by CWDA show the reasonability of this relaxation. In Appendix A [page 14], we make a further study for the necessity of normality in CWDA.


6.	About pruning for fully-connected layers (FC): In Fig.15 of Appendix D [page 18], we visualize the parameter distribution of FC. Although FC does not satisfy CWDA here, they still have the Sp that more than 0.9 between $\ell_1$, $\ell_2$, and $\mathbf{GM}$. More analysis can be found in Appendix A [page 14](about the necessity of normal distribution).

---

### Decision · Program_Chairs · 2021-01-07
**Final Decision**

**Decision:**

Reject

**Comment:**

The paper works towards analysis to understand the difference -- and primarily the lack thereof -- between different pruning methods. The central observation is that the convolutional filters in a layer are not strongly correlated and -- if the weights of the layer are taken as a matrix -- then the covariance matrix is block diagonal.

Extending this objective the regime of a large number of filters, then the matrix is approximately diagonal and all weights are -- approximately Gaussian and i.i.d. The point of this analysis is that under this assumption, norm-based metrics, particluarly $\ell_1$ and $\ell_2$, behave quite similarly.

The pros of this paper are the extensive evaluation and -- after revisions -- relatively clear text. The core analysis is nice to have elaborated in detail in the community.

The primary con of this paper is, as the reviewers point out, that there are limited conclusions to take away from this work. Specifically, a plausible default hypothesis is that different pruning criteria result in different pruning decisions. From the results in this paper, that still seems to hold with -- exception of the norm-based metrics. So, while this work does demonstrate that these norm-based metrics are relatively similar -- a nice clarification to see in the community -- the work offers limited comment on the broader space of pruning metrics.

My recommendation is Reject. Despite the strong empirical evaluation, the ultimate results offer limited clarification on the similarity of pruning metrics.